# Deep Networks and the Multiple Manifold Problem

**Sam Buchanan**
Columbia University
sdb2157@columbia.edu

**Dar Gilboa**
Harvard University
dar_gilboa@fas.harvard.edu

**John Wright**
Columbia University
jw2966@columbia.edu

## Abstract

We study the multiple manifold problem, a binary classification task modeled on applications in machine vision, in which a deep fully-connected neural network is trained to separate two low-dimensional submanifolds of the unit sphere. We provide an analysis of the one-dimensional case, proving for a simple manifold configuration that when the network depth $L$ is large relative to certain geometric and statistical properties of the data, the network width $n$ grows as a sufficiently large polynomial in $L$, and the number of i.i.d. samples from the manifolds is polynomial in $L$, randomly-initialized gradient descent rapidly learns to classify the two manifolds perfectly with high probability. Our analysis demonstrates concrete benefits of depth and width in the context of a practically-motivated model problem: the depth acts as a fitting resource, with larger depths corresponding to smoother networks that can more readily separate the class manifolds, and the width acts as a statistical resource, enabling concentration of the randomly-initialized network and its gradients. The argument centers around the "neural tangent kernel" of Jacot et al. and its role in the nonasymptotic analysis of training overparameterized neural networks; to this literature, we contribute essentially optimal rates of concentration for the neural tangent kernel of deep fully-connected ReLU networks, requiring width $n \geq L \operatorname{poly}(d_0)$ to achieve uniform concentration of the initial kernel over a $d_0$-dimensional submanifold of the unit sphere $\mathbb{S}^{n_0-1}$, and a nonasymptotic framework for establishing generalization of networks trained in the "NTK regime" with structured data. The proof makes heavy use of martingale concentration to optimally treat statistical dependencies across layers of the initial random network. This approach should be of use in establishing similar results for other network architectures.

## 1 Introduction

Data in many applications in machine learning and computer vision exhibit low-dimensional structure (Fig. 1a). Although deep neural networks achieve state-of-the-art performance on tasks in these areas, rigorous explanations for their performance remain elusive, in part due to the complex interaction between models, architectures, data, and algorithms in neural network training. There is a need for model problems that capture essential features of applications (such as low dimensionality), but are simple enough to admit rigorous end-to-end performance guarantees. In addition to helping to elucidate the mechanisms by which deep networks succeed, this approach has the potential to clarify the roles of various network properties and how these should reflect the properties of the data.

These considerations lead us to formulate the *multiple manifold problem* (Fig. 1b), a binary classification problem in which the classes are two disjoint submanifolds of the unit sphere $\mathbb{S}^{n_0-1}$, and the classifier is a deep fully-connected ReLU network of depth $L$ and width $n$ trained on $N$ i.i.d. samples from a distribution supported on the manifolds. The goal is to articulate conditions on the network architecture and number of samples under which the learned classifier *provably separates the two manifolds*, guaranteeing perfect generalization to unseen data. The difficulty of an instance of the multiple manifold problem is controlled by the dimension of the manifolds $d_0$, their separation $\Delta$, and their curvature $\kappa$, allowing us to study the constraints imposed by these intrinsic properties of the data on the settings of the neural network's architectural hyperparameters such that the two manifolds can be separated by training with a gradient-based method.

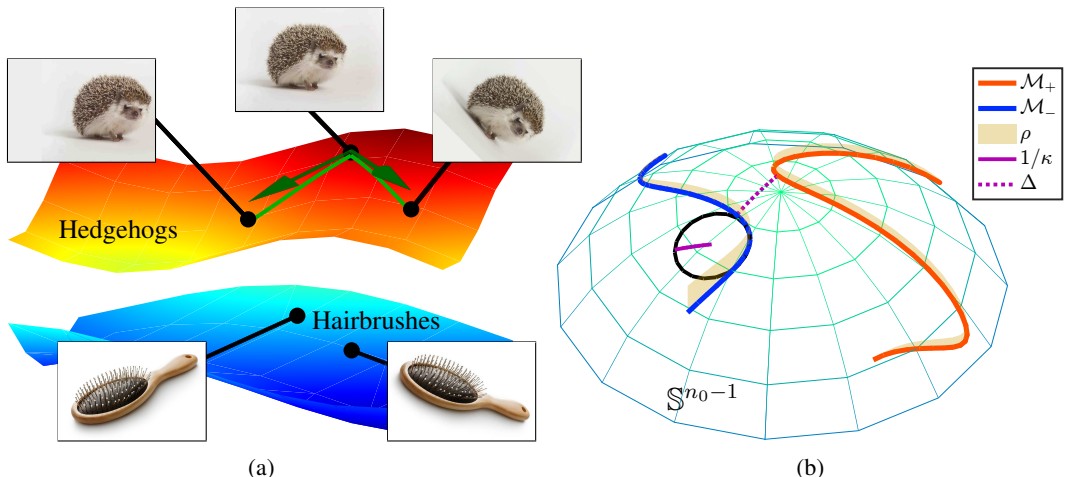

Figure 1: **(a)** Data in image classification with standard augmentation techniques, as well as other domains in which neural networks are commonly used, lies on low dimensional class manifolds—in this case those generated by the action of continuous transformations on images in the training set. Tangent vectors at a point on the manifold corresponding to an application of a rotation or a translation are illustrated in green. The dimension of the manifold is determined by the dimension of the symmetry group, and is typically small. **(b)** The *multiple manifold problem*. Our model problem, capturing this low dimensional structure, is the classification of low-dimensional submanifolds of a sphere $\mathbb{S}^{n_0-1}$. The difficulty of the problem is set by the inter-manifold separation $\Delta$ and the curvature $\kappa$. The depth and width of the network required to provably reduce the generalization error efficiently are set by these parameters.

Our main result is an analysis of the one-dimensional case of the multiple manifold problem, which reduces the analysis of the gradient descent dynamics to the construction of a *certificate*—showing that a certain deterministic integral equation involving the network architecture and the structure of the data admits a solution of small norm. We construct such a certificate for the simple geometry in Fig. 3, guaranteeing generalization in this setting.

**Theorem 1** (informal). *Let $d_0 = 1$. Suppose a certificate for $\mathcal{M}$ exists. Then if the network depth satisfies $L \geq \mathrm{poly}(\kappa, C_\rho, \log(n_0))$, the width satisfies $n \geq \mathrm{poly}(L, \log(Ln_0))$, and the number of training samples satisfies $N \geq \mathrm{poly}(L)$, randomly-initialized gradient descent on $N$ i.i.d. samples rapidly learns a network that separates the two manifolds with overwhelming probability. The constants $C_\rho, \kappa$ depend only on the data density and the regularity of the manifolds. In addition, if $L \gtrsim \Delta^{-1}$, then a certificate exists for the configuration of $\mathcal{M}$ shown in Fig. 3.*

Theorem 1 gives a provable generalization guarantee for a model classification problem with deep networks on structured data that depends *only* on the architectural hyperparameters and properties of the data. In addition, it provides an interpretable tradeoff between the architectural settings necessary to separate the two manifolds: the network depth needs to be set according to the intrinsic difficulty of the problem, and the network width needs to grow with the depth. Our analysis gives further insight into the independent roles played by each of these parameters in solving the problem, with the depth acting as a 'fitting resource', making the network's output more regular and easier to change, and the width acting as a 'statistical resource', granting concentration of the network over the random initialization around a well-behaved object that we can analyze. Moreover, the sample complexity of Theorem 1 is dictated by the intrinsic difficulty of the problem instance which is set by the geometry of the data. As a consequence, we avoid any dependence of the width of the network on the number of samples, which is common in deep network convergence results in the literature (e.g. (Allen-Zhu et al., 2019b; Du et al., 2019), (Chen et al., 2021, Theorem 3.4)). As is the case in practice, given a fixed architecture, more data doesn't have a detrimental effect on fitting [1].

Theorem 1 is modular, in the sense that a generalization guarantee is ensured for any geometry for which one can construct a certificate. The key to our approach will be to approximate the gradient

---

[1]When using data augmentation, for example, the number of samples is effectively infinite yet highly structured, enabling convergence and generalization.

descent dynamics with a linear discrete dynamical system defined in terms of the so-called neural tangent kernel $\Theta(x, x')$ defined on the manifolds. Due to the structure in the data, diagonalizing the operator corresponding to this kernel is intractable in general, but we show that constructing a certificate—arguably an easier task, because it requires producing a bound on the norm of a solution to an equation rather than producing the solution itself—suffices to guarantee that the error decreases rapidly during training given a suitably structured network.

We summarize the primary contributions of this work below.

- *Generalization in deep networks:* There are few generalization results for deep networks trained efficiently with gradient descent available in the literature.[2] Theorem 1 provides such a guarantee that does not depend on any property of the trained network (e.g., norms of final weights) that is not readily available before training. In this context, the certificate condition is equivalent to the initial network function having a controlled norm in a certain RKHS; this condition is natural in the training regime we consider, and appears ubiquitously in works on generalization in shallower networks (Ghorbani et al., 2020; Ji & Telgarsky, 2020; Nitanda & Suzuki, 2021).

- *Uniform concentration of the neural tangent kernel for deep ReLU networks:* As an intermediate step in the proof of Theorem 1, we establish essentially optimal rates of uniform concentration for the neural tangent kernel of an arbitrarily deep network (Theorem 2) using martingale concentration, where we require the width to grow only *linearly* with the depth. We expect this martingale approach to be applicable to essentially any other compositionally-structured network architecture. Our uniform result generalizes prior results on pointwise concentration (Arora et al., 2019b; Allen-Zhu et al., 2019b), analogous to our Theorem B.3, and proves useful in establishing generalization.

- *Strong regularity estimates for random ReLU networks:* As a further consequence of the uniform concentration framework we have developed, we obtain *depth-logarithmic* Lipschitz estimates for random ReLU networks of arbitrary depth and linear width, as well as (for still wider networks) a uniform approximation for the network output by a constant which improves with depth, both with overwhelming probability (Section 3.3). We also control the evolution of the Lipschitz constant during NTK regime training (Lemma B.7), showing that it scales polynomially in the depth. These results may be of interest in applications where guaranteeing a Lipschitz property for networks is important, such as GAN training (Miyato et al., 2018) or denoising (Ryu et al., 2019; Sun et al., 2020).

## 1.1 RELATED WORK

**Deep networks and low-dimensional structure.** The notion of modeling data as low-dimensional submanifolds has been widely considered in the context of clustering (Wang et al., 2015) and manifold learning (Donoho & Grimes, 2005; Fefferman et al., 2016). Goldt et al. (2020) independently proposed the "hidden manifold model", a model problem for learning shallow neural networks for binary classification of structured data with motivations very similar to ours and which admits a mean-field analysis (Gerace et al., 2020). The data model consists of gaussian samples from a low-dimensional subspace passed through a nonlinear function acting coordinatewise in the standard basis; although this models statistical variations around a base domain, a feature of real data that the model we study here lacks, we believe that the study of an arbitrary density supported on two Riemannian manifolds lends our data model increased structural generality. In the context of kernel regression with the kernel given by the NTK of a two-layer neural network, Ghorbani et al. (2020) study a data generating model that consists of uniform samples from a low-dimensional subsphere corrupted additively by independent uniform samples from a subsphere in the orthogonal complement, and a target mapping that depends only on the low-dimensional part. The authors obtain asymptotic generalization guarantees for this data model that reveal conditions under which the corruption degrades the performance of neural tangent methods.

---

[2]The closest result we are aware of is (Chen et al., 2021, Theorem 3.4); this result involves a-priori assumptions on the trained network weights, which are only resolved for two-layer networks, and entails an unnatural relationship between $n$ and $N$ and a possible exponential dependence of $N$ on $L$, which Theorem 1 avoids.

**Analyses of neural network training.** To reason analytically about the complicated training process, we adopt the neural tangent kernel approach (Jacot et al., 2018). The first works to instantiate these ideas in a nonasymptotic setting obtained convergence guarantees for training deep neural networks on finite datasets (Allen-Zhu et al., 2019b; Du et al., 2019). By exploiting more structure in the data, generalization results have been obtained (Allen-Zhu et al., 2019a; Arora et al., 2019a; Ji & Telgarsky, 2020; Oymak et al., 2019; Cao & Gu, 2019; Suzuki, 2020; Li et al., 2020; Allen-Zhu & Li, 2020) that apply to shallow networks, teacher-student learning scenarios, and/or hold conditional on the existence of certain small-norm interpolators. Other works have obtained generalization guarantees using generalization bounds for kernel methods (Ghorbani et al., 2019; Liang et al., 2020; Ghorbani et al., 2020; Montanari & Zhong, 2020) using the fact that the linearized predictor in the NTK regime can be linked to a kernel method (Arora et al., 2019b). A parallel line of works (Mei et al., 2018; Tzen & Raginsky, 2020; Mei et al., 2019; Chizat & Bach, 2020; Fang et al., 2020) approach the problem by studying an infinite-width limit of neural network training that yields a different training dynamics. Approaches of this type are of interest because there is no restriction to short-time dynamics, and the limit of the dynamics can often be characterized in terms of a well-structured object, such as a max-margin classifier (Chizat & Bach, 2020). On the other hand, it is often difficult to prove finite-time convergence to the limit.

## 2   PROBLEM FORMULATION AND MAIN RESULTS

### 2.1   DATA MODEL AND NETWORK DEFINITIONS

We consider data supported on the union of two class manifolds $\mathcal{M} = \mathcal{M}_+ \cup \mathcal{M}_-$, where $\mathcal{M}_+$ and $\mathcal{M}_-$ are two disjoint smooth regular simple curves taking values in $\mathbb{S}^{n_0-1}$, with $n_0 \geq 3$. We denote the data measure supported on $\mathcal{M}$ that generates our samples as $\mu^\infty$, with corresponding density $\rho$, and write $\rho_{\min} = \inf_{\boldsymbol{x} \in \mathcal{M}} \rho(\boldsymbol{x})$ and $\rho_{\max} = \sup_{\boldsymbol{x} \in \mathcal{M}} \rho(\boldsymbol{x})$. We denote by $\kappa$ a uniform bound on the (extrinsic) curvature of the two curves, we write $\Delta = \min_{\boldsymbol{x} \in \mathcal{M}_+, \boldsymbol{x}' \in \mathcal{M}_-} \angle(\boldsymbol{x}, \boldsymbol{x}')$ for the separation between class manifolds, where $\angle(\boldsymbol{x}, \boldsymbol{x}') = \cos^{-1}\langle \boldsymbol{x}, \boldsymbol{x}' \rangle$ for unit vectors, and to have a quantitative characterization of 'how simple' the curves are, we assume there exist constants $0 < c_\lambda \leq 1$, $K_\lambda \geq 1$ such that for every $0 < s \leq c_\lambda/\kappa$ and every $\boldsymbol{x}, \boldsymbol{x}'$ in a common connected component of $\mathcal{M}$, one has $\mathrm{dist}_\mathcal{M}(\boldsymbol{x}, \boldsymbol{x}') \leq K_\lambda s$ if $\angle(\boldsymbol{x}, \boldsymbol{x}') \leq s$, where $\mathrm{dist}_\mathcal{M}$ denotes the Riemannian distance. Our target function is the signed indicator for each class manifold $f_\star(\boldsymbol{x}) = \mathbb{1}_{\mathcal{M}_+}(\boldsymbol{x}) - \mathbb{1}_{\mathcal{M}_-}(\boldsymbol{x})$.

The model we consider is a fully-connected neural network with ReLU activations and access to i.i.d. samples from $\mu^\infty$ and their corresponding labels. We parameterize our neural network with weights $\boldsymbol{W}^1 \in \mathbb{R}^{n \times n_0}$, $\boldsymbol{W}^\ell \in \mathbb{R}^{n \times n}$ if $\ell \in \{2, \dots, L\}$, and $\boldsymbol{W}^{L+1} \in \mathbb{R}^{1 \times n}$, which we collect as $\boldsymbol{\theta} = (\boldsymbol{W}^1, \dots, \boldsymbol{W}^{L+1})$, and write the iterates of the forward pass as $\boldsymbol{\alpha}_{\boldsymbol{\theta}}^\ell(\boldsymbol{x}) = \left[ \boldsymbol{W}^\ell \boldsymbol{\alpha}_{\boldsymbol{\theta}}^{\ell-1}(\boldsymbol{x}) \right]_+$ for $\ell = 1, \dots, L$ with $\boldsymbol{\alpha}_{\boldsymbol{\theta}}^0(\boldsymbol{x}) = \boldsymbol{x}$, which we also refer to as *features* or *activations*, with the network output written as $f_{\boldsymbol{\theta}}(\boldsymbol{x}) = \boldsymbol{W}^{L+1} \boldsymbol{\alpha}_{\boldsymbol{\theta}}^L(\boldsymbol{x})$, and the prediction error as $\zeta_{\boldsymbol{\theta}}(\boldsymbol{x}) = f_{\boldsymbol{\theta}}(\boldsymbol{x}) - f_\star(\boldsymbol{x})$.

For an i.i.d. sample $(\boldsymbol{x}_1, \dots, \boldsymbol{x}_N)$ from $\mu^\infty$, we write $\mu^N = \frac{1}{N} \sum_{i=1}^N \delta_{\boldsymbol{x}_i}$ for the empirical measure associated to the sample, and we consider the training objective $\mathcal{L}_{\mu^N}(\boldsymbol{\theta}) = \frac{1}{2} \int_\mathcal{M} (\zeta_{\boldsymbol{\theta}}(\boldsymbol{x}))^2 \, \mathrm{d}\mu^N(\boldsymbol{x})$, i.e. the empirical risk evaluated with the square loss. We train with vanilla gradient descent with constant step size $\tau > 0$: after randomly initializing the parameters $\boldsymbol{\theta}_0^N$ as $\boldsymbol{W}^\ell \sim_{\mathrm{i.i.d.}} \mathcal{N}(0, 2/n)$ if $\ell \in [L]$ and $\boldsymbol{W}^{L+1} \sim_{\mathrm{i.i.d.}} \mathcal{N}(0, 1)$ we consider the sequence of iterates $\boldsymbol{\theta}_{k+1}^N = \boldsymbol{\theta}_k^N - \tau \widetilde{\nabla} \mathcal{L}_{\mu^N}(\boldsymbol{\theta}_k^N)$, where $\widetilde{\nabla} \mathcal{L}_{\mu^N}$ represents a 'formal gradient' of the empirical loss, which we define in detail in Appendix A.1. We say the parameters obtained at iteration $k$ of gradient descent *separate the manifolds* $\mathcal{M}$ if the classifier implemented by the neural network with the parameters $\boldsymbol{\theta}_k^N$ labels the two manifolds correctly, i.e. if $f_\star(\boldsymbol{x}) \mathrm{sign}(f_{\boldsymbol{\theta}_k^N}(\boldsymbol{x})) = 1$ for every $\boldsymbol{x} \in \mathcal{M}$. As a shorthand, we will denote quantities evaluated at $\boldsymbol{\theta}_k^N$ with a subscript $k$; an omitted subscript will denote $k = 0$, and we will write explicitly $\boldsymbol{\theta}_0 = \boldsymbol{\theta}_0^N$. Additional notation is provided in Appendix A.5.1.

### 2.2   ERROR DYNAMICS AND CERTIFICATES

Because it is difficult to endow the network parameters generated by the gradient iteration with a particular interpretation, we prefer to reason about how the network error $\zeta_k^N$ evolves under gradient

descent. We calculate (in Lemma B.8)

$$\zeta_{k+1}^N(\boldsymbol{x}) = \zeta_k^N(\boldsymbol{x}) - \tau \int_{\mathcal{M}} \Theta_k^N(\boldsymbol{x}, \boldsymbol{x}') \zeta_k^N(\boldsymbol{x}') \, d\mu^N(\boldsymbol{x}'), \tag{2.1}$$

where we have defined the integral kernel $\Theta_k^N(\boldsymbol{x}, \boldsymbol{x}') = \int_0^1 \langle \widetilde{\nabla} f_{\boldsymbol{\theta}_k^N}(\boldsymbol{x}'), \widetilde{\nabla} f_{\boldsymbol{\theta}_k^N - t\tau \widetilde{\nabla} \mathcal{L}_{\mu^N}(\boldsymbol{\theta}_k^N)}(\boldsymbol{x}) \rangle \, dt$,

where $\widetilde{\nabla} f_{\boldsymbol{\theta}_0}$ denotes a formal gradient of the initial network function with respect to the parameters, which is defined in detail in Appendix A.1. We then define a *nominal error evolution* by

$$\zeta_{k+1}^\infty(\boldsymbol{x}) = \zeta_k^\infty(\boldsymbol{x}) - \tau \int_{\mathcal{M}} \Theta(\boldsymbol{x}, \boldsymbol{x}') \zeta_k^\infty(\boldsymbol{x}') \, d\mu^\infty(\boldsymbol{x}') \tag{2.2}$$

with identical initial conditions $\zeta_0^\infty = \zeta$ and where $\Theta(\boldsymbol{x}, \boldsymbol{x}') = \langle \widetilde{\nabla} f_{\boldsymbol{\theta}_0}(\boldsymbol{x}), \widetilde{\nabla} f_{\boldsymbol{\theta}_0}(\boldsymbol{x}') \rangle$ is the so-called neural tangent kernel with associated integral operator $\Theta$. We prove that the error evolution (2.1) is well-approximated by the nominal error evolution under suitable conditions on the network width, step size, and number of samples, which together ensure that training proceeds in the "NTK regime" where $\Theta_k^N$ stays close to $\Theta$. As for the nominal error evolution (2.2), we note that this system is linear, time-invariant, and stable when $\tau$ is set appropriately small, so the norm of the nominal error is guaranteed to decrease rapidly if the initial error $\zeta$ aligns well with eigenfunctions of $\Theta$ corresponding to large eigenvalues. However, computation of these eigenfunctions is intractable for general data geometries and distributions because the operator $\Theta$ is not generally translationally invariant on $\mathcal{M}$. To overcome this issue, we prove this alignment *implicitly* by constructing an approximate solution to the linear integral equation $\Theta[g] = \zeta$ such that $\|g\|_{L^2_{\mu^\infty}}$ is sufficiently small. To be precise, $g \in L^2_{\mu^\infty}$ will be called a $\delta_1, \delta_2$-*certificate* for the dynamics (2.2) if

$$\|\Theta[g] - \zeta\|_{L^2_{\mu^\infty}} \le \delta_1; \quad \|g\|_{L^2_{\mu^\infty}} \le \delta_2. \tag{2.3}$$

## 2.3 MAIN RESULTS AND PROOF OUTLINE

Our main result is that conditional on the existence of a certificate of suitably small norm for $\mathcal{M}$, gradient descent provably separates the two manifolds in time polynomial in the network depth.

**Theorem 1.** *Let $\mathcal{M}$ be a one-dimensional Riemannian manifold satisfying our regularity assumptions. For any $0 < \delta \le 1/e$, choose*

$$
\begin{aligned}
L &\ge C_1 \max\{C_{\mu^\infty} \log^9(1/\delta) \log^{24}(C_{\mu^\infty} n_0 \log(1/\delta)), \kappa^2 K_\lambda^2/c_\lambda^2\}, \\
n &= C_2 L^{99} \log^9(1/\delta) \log^{18}(Ln_0), \\
N &\ge L^{10},
\end{aligned}
$$

*and fix $\tau$ such that $\frac{C_3}{nL^2} \le \tau \le \frac{C_4}{nL}$.*

*Then if there exists a certificate in the sense of (2.3) with $\delta_1 = C_5 C_\rho^{1/2} \sqrt{\log(1/\delta) \log(nn_0)}/L$ and $\delta_2 = C_6 \sqrt{\log(1/\delta) \log(nn_0)}/(n\rho_{\min}^{1/2})$, with probability at least $1 - \delta$ over the random initialization of the network and the i.i.d. sample from $\mu^\infty$, the parameters obtained at iteration $\lfloor L^{39/44}/(n\tau) \rfloor$ of gradient descent on the finite sample loss $\mathcal{L}_{\mu^N}$ yield a classifier that separates the two manifolds.*

*The constants $C_1, \ldots, C_6$ are suitably chosen absolute constants, the constants $\kappa$, $K_\lambda$, $c_\lambda$ are respectively the extrinsic curvature constant and the global regularity constant defined in Section 2.1, the constant $C_\rho$ is defined as $\max\{\rho_{\min}, \rho_{\min}^{-1}\}$, and the constant $C_{\mu^\infty}$ is defined as $C_\rho^{15}(1 + \rho_{\max})^6 (\min\{\mu^\infty(\mathcal{M}_+), \mu^\infty(\mathcal{M}_-)\})^{-11/2}$.*

For one-dimensional instances of the two manifold problem with sufficiently deep and overparameterized networks trained in the small-step-size regime, Theorem 1 completely reduces the analysis of the gradient iteration to the certificate problem. From a qualitative perspective, the network resource constraints imposed by Theorem 1 are natural:

(i) The network depth $L$ is set by geometric and statistical properties of the data with only a mild polylogarithmic dependence on the ambient dimension $n_0$, which reflects the role of depth in controlling the capability of the network to fit functions.

(ii) The network width $n$ is set by the depth $L$: the inductive structure of the network causes quantities that depend on the initial random weights $\boldsymbol{\theta}_0$ to concentrate worse as the depth is increased, which can be counteracted by setting the width appropriately large.

(iii) The sample complexity of $N \geq L^{10}$ reflects the capacity of the network via the depth, and is in particular independent of the width $n$, which can thus be interpreted as purely a statistical resource.

In addition, the conclusion of Theorem 1 implies not just that the expected generalization error with respect to $\mu^\infty$ of a binary classifier is zero, but the stronger separation property, i.e. that the generalization error will be zero *for any choice of test distribution supported on* $\mathcal{M}$ simultaneously. We give a brief sketch of the proof of Theorem 1 in Appendix A.4. To obtain a generalization guarantee from Theorem 1, it only remains to construct a certificate for $\mathcal{M}$. We demonstrate this for the family of simple, highly-symmetric geometries shown in Figure 3, and leave the case of general one-dimensional manifolds for future work.

**Proposition 1.** *Let $\mathcal{M}$ be an $r$-instance of the two circles geometry studied in Appendix C.1.1 and shown in Figure 3, with $r \geq 1/2$. For any $0 < \delta \leq 1/e$, if $L \geq C_1\Delta^{-1}$ and $n \geq C_2L^5\log^4(1/\delta)\log^4(Ln_0\log(1/\delta))$, then there exists a certificate in the sense of (2.3) satisfying the requirements of Theorem 1 with probability at least $1 - 3\delta$ for some absolute constants $C_1, C_2 > 0$.*

Taking a union bound, Proposition 1 shows that under the hypotheses of Theorem 1, with probability at least $1 - 4\delta$ a certificate exists for the geometry shown in Figure 3 as soon as $L$ is larger than a constant multiple of the inverse separation $\Delta^{-1}$, even as the separation approaches zero. We conjecture that a similar phenomenon holds for more general geometries, possibly with additional dependencies on the curvature and global regularity parameters of $\mathcal{M}$. The dependence of $L$ on the geometry is due to the "sharpening" effect the depth has on the kernel $\Theta$ governing the dynamics and thus on the fitting capabilities of the network, as illustrated in Figure 2a.

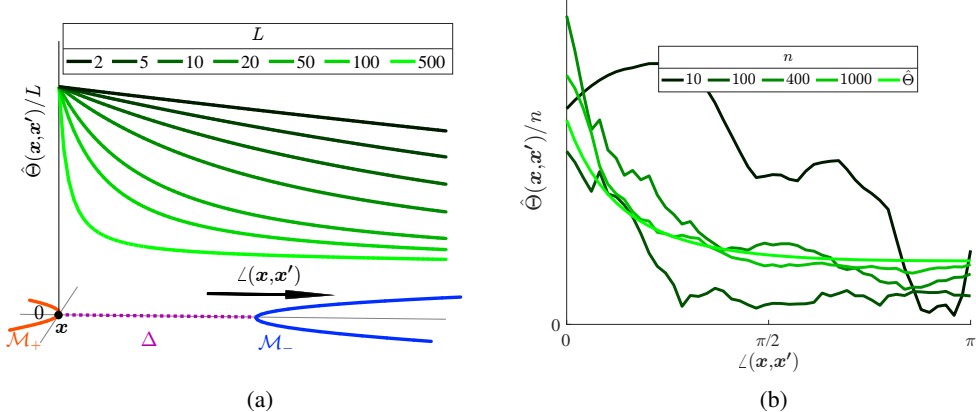

(a)                                              (b)

Figure 2: **(a)** *Depth acts as a fitting resource.* As $L$ increases, the rotationally-invariant kernel $\hat{\Theta}$ (a slight modification of the deterministic kernel in Theorem 2) decays more rapidly as a function of angle between the inputs $\angle(\boldsymbol{x}, \boldsymbol{x}')$ ($n$ is held constant). Below the curves we show an isometric chart around a point $\boldsymbol{x} \in \mathcal{M}_+$. Once the decay scale of $\hat{\Theta}$ is small compared to the inter-manifold distance $\Delta$ and the curvature of $\mathcal{M}_-$, the network output can be changed at $\boldsymbol{x}$ while only weakly affecting its value on $\mathcal{M}_-$. This is one mechanism that relates the depth required to solve the classification problem to the data geometry. **(b)** *Width acts as a statistical resource.* The dynamics at initialization are governed by $\Theta$, a random process over the network parameters. As $n$ is increased, the normalized fluctuations of $\Theta$ around $\hat{\Theta}$ decrease (here $L = 10$). These two phenomena are related, since the fluctuations also grow with depth, as evinced by the scaling in Theorem 2.

To prove that the nominal error evolution (2.2) decreases rapidly and approximates the actual error evolution (2.1) throughout training, it is essential to have a precise characterization of the 'initial' neural tangent kernel $\Theta$. One of our main technical contributions is to show concentration of $\Theta$ in the regime where the width $n$ scales *linearly* with the depth $L$.

**Theorem 2.** *For any $d_0 \in \mathbb{N}$, let $\mathcal{M}$ be a $d_0$-dimensional complete Riemannian submanifold of $\mathbb{S}^{n_0-1}$. Then if $n \geq C_1 L d_0^4 \log^4(C_\mathcal{M} n_0 L)$, one has with probability at least $1 - n^{-10}$ that for every $(\boldsymbol{x}, \boldsymbol{x}') \in \mathcal{M} \times \mathcal{M}$*

$$\left| \Theta(\boldsymbol{x}, \boldsymbol{x}') - \frac{n}{2} \sum_{\ell=0}^{L-1} \cos\left(\varphi^{(\ell)}(\nu)\right) \prod_{\ell'=\ell}^{L-1} \left(1 - \frac{\varphi^{(\ell')}(\nu)}{\pi}\right) \right| \leq C_2 \sqrt{nL^3 d_0^4 \log^4(C_\mathcal{M} n n_0)},$$

*where we write $\nu = \angle(\boldsymbol{x}, \boldsymbol{x}')$ with an abuse of notation, $\varphi^{(\ell)}$ denotes the $\ell$-fold composition of $\varphi(\nu) = \cos^{-1}\left((1 - \frac{\nu}{\pi}) \cos \nu + \frac{\sin \nu}{\pi}\right)$, the constants $C_1, C_2 > 0$ are absolute, and the constant $C_\mathcal{M} > 0$ depends only on the diameters and curvatures of the class manifolds (Lemma C.4).* [3]

For networks of uniform width that are wider than they are deep by a certain constant factor, we believe that the scalings in Theorem 2 are essentially optimal: the variance calculations of Hanin & Nica (2020) give some heuristic evidence here, and we believe the idea of using diagonal concentration to prove deviation lower bounds could be generalized to rigorously establish optimality. Figure 2b illustrates the phenomenon underlying Theorem 2. We discuss the proof of Theorem 2 in more detail in Sections 3.1 and 3.3.

## 3  Key Proof Elements

### 3.1  Concentration at Initialization: Martingales and Angle Contraction

The initial kernel $\Theta$ is a complicated random process defined over the weights $(\boldsymbol{W}^1, \ldots, \boldsymbol{W}^{L+1})$. To control it, we first show for fixed $(\boldsymbol{x}, \boldsymbol{x}')$ that $\Theta(\boldsymbol{x}, \boldsymbol{x}')$ concentrates with high probability, and then leverage approximate continuity properties to pass to uniform control of $\Theta$. Here we describe our approach to pointwise control; uniformization is discussed in Section 3.3. The kernel can be written in the form

$$\Theta(\boldsymbol{x}, \boldsymbol{x}') = \langle \boldsymbol{\alpha}^L(\boldsymbol{x}), \boldsymbol{\alpha}^L(\boldsymbol{x}') \rangle + \sum_{\ell=0}^{L-1} \langle \boldsymbol{\alpha}^\ell(\boldsymbol{x}), \boldsymbol{\alpha}^\ell(\boldsymbol{x}') \rangle \langle \boldsymbol{\beta}^\ell(\boldsymbol{x}), \boldsymbol{\beta}^\ell(\boldsymbol{x}') \rangle,$$

where $\boldsymbol{\beta}^\ell(\boldsymbol{x}) = (\boldsymbol{W}^{L+1} \boldsymbol{P}_{I_L(\boldsymbol{x})} \cdots \boldsymbol{W}^{\ell+2} \boldsymbol{P}_{I_{\ell+1}(\boldsymbol{x})})^*$ will be referred to as *backward features*, and $\boldsymbol{P}_{I_\ell(\boldsymbol{x})}$ is a projection onto $\{\boldsymbol{\alpha}^\ell(\boldsymbol{x}) > \boldsymbol{0}\}$. We consider $\langle \boldsymbol{\beta}^0(\boldsymbol{x}), \boldsymbol{\beta}^0(\boldsymbol{x}') \rangle$ as a representative example: up to a small residual term, this random variable can be expressed as a sum of martingale differences. Formally, for $\ell \in [L]$, let $\mathcal{F}^\ell$ denote the $\sigma$-algebra generated by all weight matrices up to layer $\ell$, with $\mathcal{F}^0$ denoting the trivial $\sigma$-algebra. We can then write

$$\left| \langle \boldsymbol{\beta}^0(\boldsymbol{x}), \boldsymbol{\beta}^0(\boldsymbol{x}') \rangle - g_0(\nu^0) \right| \leq \left| \sum_{\ell=1}^{L+1} g_\ell(\boldsymbol{W}^\ell, \ldots, \boldsymbol{W}^1, \nu^0) - \mathbb{E}\left[ g_\ell(\boldsymbol{W}^\ell, \ldots, \boldsymbol{W}^1, \nu^0) \,\middle|\, \mathcal{F}^{\ell-1} \right] \right| + R$$
(3.1)

for some functions $g_\ell$ and controllable residual $R$, where $\nu^0 = \angle(\boldsymbol{x}, \boldsymbol{x}')$. If we fix all the variables in $\mathcal{F}^{\ell-1}$, the fluctuations in the $\ell$-th summand will be due to $\boldsymbol{W}^\ell$ alone. Intuitively, since each weight matrix appears at most once in $\boldsymbol{\beta}^0(\boldsymbol{x})$,[4] it will appear at most twice in $g_\ell$, and therefore $g_\ell$ will have a subexponential distribution conditioned on $\mathcal{F}^{\ell-1}$ and concentrate well around its conditional expectation. This property stems from the compositional structure of the network, with independent sources of randomness introduced at every layer, and is essentially agnostic to other details of the architecture. The concentration of the summands in (3.1) implies concentration of the sum: even though the summands are not independent, they can be controlled using concentration inequalities analogous to those for sums of independent variables (Azuma, 1967; Freedman, 1975).

Showing that terms of the form $\langle \boldsymbol{\alpha}^\ell(\boldsymbol{x}), \boldsymbol{\alpha}^\ell(\boldsymbol{x}') \rangle$ concentrate in the linear regime gives rise to additional challenges. Here we exploit an essential difference between the concentration properties

---

[3]Since we do not use the "NTK parameterization", the norm of our NTK scales like $nL$ rather than $L$. Due to our scaling of the weights (Section 2.1) the contribution of the final layer to the NTK is negligible and can be dropped. This leads to discrepancies between the expression above and similar expressions found in the literature—we show essential equivalence between our NTK and others in Appendix A.3.

[4]Technically, the features $\boldsymbol{\alpha}^\ell(\boldsymbol{x})$ depend on all the weights up to layer $\ell$ and hence so does the projection matrix $\boldsymbol{P}_{I_\ell(\boldsymbol{x})}$, but our analysis shows that this dependence has only a minor effect on the statistical fluctuations.

of the angles between features $\nu^\ell = \angle(\boldsymbol{\alpha}^\ell(\boldsymbol{x}), \boldsymbol{\alpha}^\ell(\boldsymbol{x}'))$ relative to those of the correlation process $\langle \boldsymbol{\alpha}^\ell(\boldsymbol{x}), \boldsymbol{\alpha}^\ell(\boldsymbol{x}') \rangle$ studied in prior works on concentration of $\Theta$: when $\nu^{\ell-1} = 0$, we have that $\nu^\ell = 0$ *deterministically*, whereas the correlation process behaves like a subexponential random variable with small but nonzero deviations. Together with smoothness, this clamping phenomenon allows us to show concentration of the angle at layer $\ell$ around the function $\varphi^{(\ell)}(\nu^0)$, which is no larger than a constant multiple of $\ell^{-1}$. This contraction of the angles with depth is the key to establishing Theorem 2; in addition, it gives the invariant kernel $\hat{\Theta}$ (see Figure 2b) its sharpness at zero and localization properties, both of which increase as the depth is increased and which we exploit in the proof of Proposition 1. We provide full details of our approach in Appendices D and E.

## 3.2 CERTIFICATE CONSTRUCTION: GENERAL FORMULATION AND A SIMPLE EXAMPLE

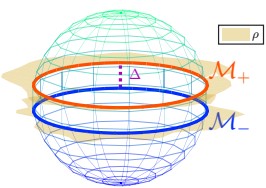

By a simple argument that relies on positiveness of $\Theta$, we show that if we can solve the certificate problem (2.3), then for a suitably chosen learning rate $\tau$ and number of iterations $k$ (Lemma B.6)

$$\mathbb{P}\left[\|\zeta_k^\infty\|_{L_{\mu^\infty}^2} \leq C_\rho \frac{\sqrt{d}\log L}{L}\right] \geq 1 - e^{-cd}.$$

Figure 3: The coaxial circles geometry.

If the network is sufficiently deep, the norm of the nominal error can thus be made arbitrarily small in a number of iterations that scales only polynomially with the problem parameters.

Because our formulation of the certificate problem (2.3) accommodates approximate solutions, under a minor condition on the network width $n$ (see Proposition 1) it suffices to solve an auxiliary system $\hat{\Theta}[g] = \hat{\zeta}$, where $\hat{\Theta}$ and $\hat{\zeta}$ are analytically-convenient approximations to $\Theta$ and $\zeta$ produced by our concentration analysis, including Theorem 2. For the simple geometry in Fig. 3, we show in Appendix C.1.1 how to solve this auxiliary system using Fourier analysis, where we require $L \gtrsim \Delta^{-1}$. The depth of the network is thus determined by the geometry of the data, and specifically by the inter-manifold distance which intuitively sets the "difficulty" of the fitting problem. In Section 4 we discuss approaches to constructing certificates for general smooth curves.

## 3.3 UNIFORM CONCENTRATION AND ITS CONSEQUENCES

To uniformize the pointwise estimates of Section 3.1, we must overcome the issue that the backward features $\boldsymbol{\beta}^\ell(\boldsymbol{x})$ are not continuous functions of the input, due to the matrices $\boldsymbol{P}_{I_\ell(\boldsymbol{x})}$. Our approach is to discretize the input space, control the number of features that can change sign near each point in the discretization, then extend the pointwise estimates of Section 3.1 to the setting where a small number of features have changed sign—again, we find martingale concentration a necessity to achieve linear width-depth scaling. We give full details in Appendix D.3.

Although Theorem 2 is the main application of these estimates—with uniform control of $\Theta$, we can prove operator norm bounds on its corresponding integral operator $\Theta$, which is of great help in proving generalization results—they also imply useful regularity estimates for the initial random network $f_{\boldsymbol{\theta}_0}$. For example, we prove that networks of uniform width $n \asymp n_0^4 L$ are with high probability $\sqrt{n_0(\log n_0)(\log L)}$-Lipschitz as functions on $\mathbb{R}^{n_0}$ (Theorem B.5)—in particular, the Lipschitz constant depends only logarithmically on depth, in contrast to existing results in the literature (Nguyen et al., 2020). For networks of larger width $n \gtrsim d_0^3 L^5$, we prove that with high probability the network $f_{\boldsymbol{\theta}_0}$ is *approximately constant* on the domain $\mathcal{M} \subset \mathbb{S}^{n_0-1}$ (Lemma D.11):

$$\sup_{\boldsymbol{x} \in \mathcal{M}} \left| f_{\boldsymbol{\theta}_0}(\boldsymbol{x}) - \int_{\mathcal{M}} f_{\boldsymbol{\theta}_0}(\boldsymbol{x}') \, \mathrm{d}\mu^\infty(\boldsymbol{x}') \right| \lesssim \frac{1}{L}.$$

We find this result to be useful in simplifying the certificate construction problem of Section 3.2.

## 4 DISCUSSION

**Certificates for curves.** The most urgent task toward expanding the scope of Theorem 1 is the construction of certificates for geometries beyond the coaxial circles of Proposition 1. The proof

of Proposition 1 relies heavily on translation invariance of the intra- and inter-manifold distances in the coaxial circles geometry in order to avoid the need for certain sharp technical estimates for the decay of the kernel $\hat{\Theta}$. With sharper control of the decay of the kernel $\hat{\Theta}$, it is possible to select the network depth in a way that grants appropriate worst-case control of the magnitude of the cross-manifold integrals in the action of $\hat{\Theta}$ (as in Figure 2a), allowing us to reduce to what is essentially a one-manifold certificate construction problem that can be solved with harmonic analysis. Beyond these considerations, it is important to extend Theorem 1 to manifolds of dimension $d_0 > 1$, which should be relatively straightforward. Our concentration results, notably including Theorem 2, are already applicable to manifolds of arbitrary dimension.

**Convolutional networks and non-differentiable manifolds.** Although we have motivated our data model in the multiple manifolds problem using applications in computer vision, it is important to note that the spatially-structured *image articulation manifolds* that arise as data in these contexts do not carry a differentiable structure (Wakin et al., 2005), so the assumption of bounded curvature may not be realistic here. On the other hand, in these applications it is standard to employ a convolutional network architecture. We anticipate that our martingale concentration framework can be extended to these architectures, and beyond establishing analogues of Theorem 1 in this setting, we believe it should be possible to obtain similar guarantees for models of image articulation manifolds. In particular, one might expect randomly-initialized convolutional networks to enjoy *local* invariance properties, like the scattering networks of Mallat (Mallat, 2012; Bruna & Mallat, 2013), which could achieve a degree of invariant classification without expending additional network resources computing convolutions over general LCA groups (Cohen & Welling, 2016).

**The importance of being low-dimensional.** Ghorbani et al. (2019) show that kernel ridge regression with any rotationally invariant kernel on $\mathbb{S}^d$ (including that of a deep network) is equivalent to polynomial regression with a degree $p$ polynomial if the number of samples is bounded by $d^{p+1}$ and $d \to \infty$. For data lying on a low-dimensional manifold, as we consider here, one would expect less pessimistic rates; indeed, in a subsequent work (Ghorbani et al., 2020) the authors establish similar guarantees to Ghorbani et al. (2019) for a linear data model with low-dimensional structure in terms of a smaller "effective dimension". In comparison, although our present certificate construction argument only implies dynamics for the restrictive coaxial circles geometry of Figure 3, for which one can obtain guarantees for kernel regression with a shallow NTK by the results of Ghorbani et al. (2020), the general multiple manifold problem formulation allows one to model *nonlinear* structure in the data, and measures fitting difficulty through intrinsic parameters like the curvature and separation. The guarantees in Ghorbani et al. (2019; 2020) depend on the degree of approximability of the target function by low-degree polynomials, and although this achieves additional generality over our model, it seems more challenging to relate this to geometric or other types of nonlinear low-dimensional structure.

**The NTK regime and beyond.** In recent years there has been much work devoted to the analysis of networks trained in the regime where the changes in $\Theta_k^N$ remain small and the dynamics in (2.1) are close to linear (Jacot et al., 2018; Lee et al., 2019; Arora et al., 2019b; Allen-Zhu & Li, 2019) (referred to as the NTK/"overparametrized"/kernel regime). Concurrently, there have also been results highlighting the limitations of this regime. In (Chizat & Bach, 2018) the authors coin the term "lazy training" in referring to dynamics where the relative change in the differential of the network function is small compared to the change in the objective during gradient descent. While the dynamics we study indeed fall into this category, the analysis makes it evident that not all lazy training regimes are created equal. Our performance guarantees depend on the structure of the kernel $\hat{\Theta}$, and on controlling the fluctuations of $\Theta_k^N$ around it. We are able to control these only if the width of the network is sufficiently large compared to the depth. In contrast, lazy training can also be achieved in homogeneous models by simply scaling the output of the model (Chizat & Bach, 2018), in which case one cannot argue that the kernel has the decay properties that enable it to fit data.

Our analysis hinges on staying in the NTK regime during training. We obtain suboptimal scaling of $n$ with $L$ in Theorem 1 because we treat all changes that occur in $\Theta_k^N$ during training as being *adversarial* to the algorithm's ability to generalize. It is likely that if an improved understanding of feature learning can be incorporated into an analysis of the dynamics, the resulting scaling requirements would be more realistic.

ACKNOWLEDGMENTS

This work was supported by the grants NSF 1733857, NSF 1838061, NSF 1740833, NSF 1740391, NSF NeuroNex Award DBI-1707398 (DG), the Gatsby Charitable Foundation (DG) and a Swartz fellowship (DG), and by a fellowship award (SB) through the National Defense Science and Engineering Graduate (NDSEG) Fellowship Program, sponsored by the Air Force Research Laboratory (AFRL), the Office of Naval Research (ONR) and the Army Research Office (ARO). The authors would like to thank Ethan Dyer, Guy Gur-Ari, Quynh Nguyen, Jeffrey Pennington, Sam Schoenholz, Daniel Soudry, and Tingran Wang for helpful discussions/feedback.

REFERENCES

P-A Absil, R Mahony, and R Sepulchre. *Optimization Algorithms on Matrix Manifolds*. Princeton University Press, April 2009.

Zeyuan Allen-Zhu and Yuanzhi Li. What can resnet learn efficiently, going beyond kernels? In *Advances in Neural Information Processing Systems*, pp. 9015–9025, 2019.

Zeyuan Allen-Zhu and Yuanzhi Li. Backward feature correction: How deep learning performs deep learning. *arXiv preprint arXiv:2001.04413*, January 2020.

Zeyuan Allen-Zhu, Yuanzhi Li, and Yingyu Liang. Learning and generalization in overparameterized neural networks, going beyond two layers. In *Advances in Neural Information Processing Systems*, volume 32, 2019a.

Zeyuan Allen-Zhu, Yuanzhi Li, and Zhao Song. A convergence theory for deep learning via Over-Parameterization. In *Proceedings of the 36th International Conference on Machine Learning*, volume 97 of *Proceedings of Machine Learning Research*, pp. 242–252. PMLR, 2019b.

Sanjeev Arora, Simon Du, Wei Hu, Zhiyuan Li, and Ruosong Wang. Fine-Grained analysis of optimization and generalization for overparameterized Two-Layer neural networks. In *Proceedings of the 36th International Conference on Machine Learning*, volume 97 of *Proceedings of Machine Learning Research*, pp. 322–332. PMLR, 2019a.

Sanjeev Arora, Simon S Du, Wei Hu, Zhiyuan Li, Russ R Salakhutdinov, and Ruosong Wang. On exact computation with an infinitely wide neural net. In *Advances in Neural Information Processing Systems*, pp. 8139–8148, 2019b.

Kazuoki Azuma. Weighted sums of certain dependent random variables. *Tohoku Mathematical Journal, Second Series*, 19(3):357–367, 1967.

Stéphane Boucheron, Maud Thomas, et al. Concentration inequalities for order statistics. *Electronic Communications in Probability*, 17, 2012.

Stéphane Boucheron, Gábor Lugosi, and Pascal Massart. *Concentration Inequalities: A Nonasymptotic Theory of Independence*. OUP Oxford, February 2013.

Haim Brezis. *Functional Analysis, Sobolev Spaces and Partial Differential Equations*. Springer, New York, NY, 2011.

Joan Bruna and Stéphane Mallat. Invariant scattering convolution networks. *IEEE Trans. Pattern Anal. Mach. Intell.*, 35(8):1872–1886, August 2013.

Yuan Cao and Quanquan Gu. Generalization bounds of stochastic gradient descent for wide and deep neural networks. In *Advances in Neural Information Processing Systems*, volume 32, 2019.

Zixiang Chen, Yuan Cao, Difan Zou, and Quanquan Gu. How much over-parameterization is sufficient to learn deep ReLU networks? In *International Conference on Learning Representations*, 2021.

Lénaïc Chizat and Francis Bach. A note on lazy training in supervised differentiable programming. *CoRR*, abs/1812.07956, 2018.

Lénaïc Chizat and Francis Bach. Implicit bias of gradient descent for wide two-layer neural networks trained with the logistic loss. In *Proceedings of Thirty Third Conference on Learning Theory*, volume 125 of *Proceedings of Machine Learning Research*, pp. 1305–1338. PMLR, 2020.

Youngmin Cho and Lawrence K Saul. Kernel methods for deep learning. In *Advances in neural information processing systems*, pp. 342–350, 2009.

Taco Cohen and Max Welling. Group equivariant convolutional networks. In *Proceedings of The 33rd International Conference on Machine Learning*, volume 48 of *Proceedings of Machine Learning Research*, pp. 2990–2999, New York, New York, USA, 2016. PMLR.

Donald L Cohn. *Measure Theory*. Birkhäuser, New York, NY, 2 edition, 2013.

R M Corless, G H Gonnet, D E G Hare, D J Jeffrey, and D E Knuth. On the LambertW function. *Adv. Comput. Math.*, 5(1):329–359, December 1996.

Herbert A. David. *Order Statistics*, pp. 1039–1040. Springer Berlin Heidelberg, Berlin, Heidelberg, 2011. ISBN 978-3-642-04898-2.

Victor H de la Peña. A general class of exponential inequalities for martingales and ratios. *Ann. Probab.*, 27(1):537–564, January 1999.

David L Donoho and Carrie Grimes. Image manifolds which are isometric to euclidean space. *J. Math. Imaging Vis.*, 23(1):5–24, July 2005.

Simon S Du, Xiyu Zhai, Barnabas Poczos, and Aarti Singh. Gradient descent provably optimizes over-parameterized neural networks. In *International Conference on Learning Representations*, 2019.

Lawrence Craig Evans and Ronald F Gariepy. *Measure Theory and Fine Properties of Functions*. CRC Press, December 1991.

Cong Fang, Jason D Lee, Pengkun Yang, and Tong Zhang. Modeling from features: a mean-field framework for over-parameterized deep neural networks. *arXiv preprint arXiv:2007.01452*, July 2020.

Charles Fefferman, Sanjoy Mitter, and Hariharan Narayanan. Testing the manifold hypothesis. *J. Amer. Math. Soc.*, 29(4):983–1049, February 2016.

David A Freedman. On tail probabilities for martingales. *Ann. Probab.*, 3(1):100–118, February 1975.

Federica Gerace, Bruno Loureiro, Florent Krzakala, Marc Mezard, and Lenka Zdeborova. Generalisation error in learning with random features and the hidden manifold model. In *Proceedings of the 37th International Conference on Machine Learning*, volume 119 of *Proceedings of Machine Learning Research*, pp. 3452–3462. PMLR, 2020.

Behrooz Ghorbani, Song Mei, Theodor Misiakiewicz, and Andrea Montanari. Linearized two-layers neural networks in high dimension. *CoRR*, abs/1904.12191, 2019.

Behrooz Ghorbani, Song Mei, Theodor Misiakiewicz, and Andrea Montanari. When do neural networks outperform kernel methods? In *Advances in Neural Information Processing Systems*, volume 33, pp. 14820–14830. Curran Associates, Inc., 2020.

Sebastian Goldt, Marc Mézard, Florent Krzakala, and Lenka Zdeborová. Modeling the influence of data structure on learning in neural networks: The hidden manifold model. *Phys. Rev. X*, 10(4): 041044, December 2020.

Boris Hanin and Mihai Nica. Finite depth and width corrections to the neural tangent kernel. In *International Conference on Learning Representations*, 2020.

Christopher Heil. *A Basis Theory Primer: Expanded Edition*. Birkhäuser Boston, 2011.

Roger A Horn, Roger A Horn, and Charles R Johnson. *Topics in matrix analysis*. Cambridge university press, 1994.

Arthur Jacot, Franck Gabriel, and Clement Hongler. Neural tangent kernel: Convergence and generalization in neural networks. In *Advances in Neural Information Processing Systems*, volume 31. Curran Associates, Inc., 2018.

Ziwei Ji and Matus Telgarsky. Polylogarithmic width suffices for gradient descent to achieve arbitrarily small test error with shallow ReLU networks. In *International Conference on Learning Representations*, 2020.

Michel Ledoux and Michel Talagrand. *Probability in Banach Spaces: Isoperimetry and Processes*. Springer, Berlin, Heidelberg, 1991.

Jaehoon Lee, Lechao Xiao, Samuel Schoenholz, Yasaman Bahri, Roman Novak, Jascha Sohl-Dickstein, and Jeffrey Pennington. Wide neural networks of any depth evolve as linear models under gradient descent. In *Advances in Neural Information Processing Systems*, volume 32. Curran Associates, Inc., 2019.

John M Lee. *Introduction to Riemannian Manifolds*. Springer, Cham, 2 edition, 2018.

Yuanzhi Li, Tengyu Ma, and Hongyang R Zhang. Learning Over-Parametrized Two-Layer neural networks beyond NTK. In *Proceedings of Thirty Third Conference on Learning Theory*, volume 125 of *Proceedings of Machine Learning Research*, pp. 2613–2682. PMLR, 2020.

Tengyuan Liang, Alexander Rakhlin, and Xiyu Zhai. On the multiple descent of Minimum-Norm interpolants and restricted lower isometry of kernels. In *Proceedings of Thirty Third Conference on Learning Theory*, volume 125 of *Proceedings of Machine Learning Research*, pp. 2683–2711. PMLR, 2020.

Stéphane Mallat. Group invariant scattering. *Commun. Pure Appl. Math.*, 65(10):1331–1398, October 2012.

Song Mei, Andrea Montanari, and Phan-Minh Nguyen. A mean field view of the landscape of two-layer neural networks. *Proc. Natl. Acad. Sci. U. S. A.*, 115(33):E7665–E7671, August 2018.

Song Mei, Theodor Misiakiewicz, and Andrea Montanari. Mean-field theory of two-layers neural networks: dimension-free bounds and kernel limit. In *Proceedings of the Thirty-Second Conference on Learning Theory*, volume 99 of *Proceedings of Machine Learning Research*, pp. 2388–2464, Phoenix, USA, 2019. PMLR.

Takeru Miyato, Toshiki Kataoka, Masanori Koyama, and Yuichi Yoshida. Spectral normalization for generative adversarial networks. In *International Conference on Learning Representations*, 2018.

Andrea Montanari and Yiqiao Zhong. The interpolation phase transition in neural networks: Memorization and generalization under lazy training. *arXiv preprint arXiv:2007.12826*, July 2020.

Robb J Muirhead. *Aspects of Multivariate Statistical Theory*. Wiley Series in Probability and Statistics. John Wiley & Sons, Inc., Hoboken, NJ, USA, March 1982.

Quynh Nguyen, Marco Mondelli, and Guido Montufar. Tight bounds on the smallest eigenvalue of the neural tangent kernel for deep ReLU networks. *arXiv preprint arXiv:2012.11654*, December 2020.

Atsushi Nitanda and Taiji Suzuki. Optimal rates for averaged stochastic gradient descent under neural tangent kernel regime. In *International Conference on Learning Representations*, 2021.

Samet Oymak, Zalan Fabian, Mingchen Li, and Mahdi Soltanolkotabi. Generalization guarantees for neural networks via harnessing the low-rank structure of the jacobian. *arXiv preprint arXiv:1906.05392*, June 2019.

Mark Rudelson and Roman Vershynin. Non-asymptotic theory of random matrices: Extreme singular values. In *Proceedings of the International Congress of Mathematicians 2010 (ICM 2010)*, pp. 1576–1602. June 2011.

Ernest Ryu, Jialin Liu, Sicheng Wang, Xiaohan Chen, Zhangyang Wang, and Wotao Yin. Plug-and-Play methods provably converge with properly trained denoisers. In *Proceedings of the 36th International Conference on Machine Learning*, volume 97 of *Proceedings of Machine Learning Research*, pp. 5546–5557. PMLR, 2019.

John M Sullivan. Curves of finite total curvature. In *Discrete differential geometry*, pp. 137–161. Springer, 2008.

Y Sun, J Liu, and U S Kamilov. Block coordinate regularization by denoising. *IEEE Transactions on Computational Imaging*, 6:908–921, 2020.

Taiji Suzuki. Generalization bound of globally optimal non-convex neural network training: Transportation map estimation by infinite dimensional langevin dynamics. In *Advances in Neural Information Processing Systems*, volume 33, pp. 19224–19237. Curran Associates, Inc., 2020.

Michel Talagrand. Concentration of measure and isoperimetric inequalities in product spaces. *Publications Mathématiques de l'Institut des Hautes Etudes Scientifiques*, 81(1):73–205, 1995.

Belinda Tzen and Maxim Raginsky. A mean-field theory of lazy training in two-layer neural nets: entropic regularization and controlled McKean-Vlasov dynamics. *arXiv preprint arXiv:2002.01987*, February 2020.

Roman Vershynin. *High-dimensional probability: An introduction with applications in data science*, volume 47. Cambridge University Press, 2018.

Michael B Wakin, David L Donoho, Hyeokho Choi, and Richard G Baraniuk. The multiscale structure of non-differentiable image manifolds. In *Wavelets XI*, volume 5914, pp. 59141B. International Society for Optics and Photonics, 2005.

Xu Wang, Konstantinos Slavakis, and Gilad Lerman. Multi-Manifold Modeling in Non-Euclidean spaces. In *Proceedings of the Eighteenth International Conference on Artificial Intelligence and Statistics*, volume 38 of *Proceedings of Machine Learning Research*, pp. 1023–1032, San Diego, California, USA, 2015. PMLR.

Jonathan Weed and Francis Bach. Sharp asymptotic and finite-sample rates of convergence of empirical measures in wasserstein distance. *Bernoulli*, 25(4A):2620–2648, November 2019.

Shunhui Zhu. The comparison geometry of ricci curvature. In *Comparison Geometry*, volume 30 of *MSRI Publications*, pp. 221–262. Cambridge University Press, May 1997.

CONTENTS

## APPENDICES: SUMMARY OF CONTENTS

We briefly summarize the contents of each of the subsequent appendices.

A. We discuss the contents of the problem formulation section from the main body, Section 2.1, in more technical detail, in particular giving technical definitions for formal gradients, regularity conditions, and so on. We provide a proof sketch to offer some intuitions about the proof of the main result. We also summarize notation and the key operator definitions that appear throughout the paper.

B. We give proofs for our main results. We provide supporting results on the NTK regime dynamics of gradient descent and other relevant technical lemmas, as discussed in the proof sketch of Section 2.3.

C. We give technical definitions relevant to the cross-manifold perspective on certificate construction, construct a certificate for the two circles geometry of Figure 3, and provide technical estimates on the kernels $\psi_1$ and $\psi$ that remain after applying our measure concentration arguments to the NTK $\Theta$.

D. We collect results on measure concentration relevant to proving our main uniform concentration result for the NTK, Theorem 2. Some of these results are also relevant for controlling changes during training.

E. We collect results relevant to proving a certain concentration result for the angles between features as they propagate across one layer of the initial neural network. The main results of this section are fundamental to the study of the concentration of angles in Appendix D, and we provide them in a separate appendix due to their length.

F. We establish results on uniform control of the changes during training of the NTK $\Theta_k^N$ from its "initial value" of $\Theta$. These are a key ingredient in our dynamics arguments in Appendix B.

G. We provide statements of general technical lemmas that are of a classical nature, which we rely on throughout the other appendices.

## A   EXTENDED PROBLEM FORMULATION

### A.1   REGARDING THE ALGORITHM

We analyze a gradient-like method for the minimization of the empirical loss $\mathcal{L}_{\mu^N}$. After randomly initializing the parameters $\boldsymbol{\theta}_0^N$ as $\boldsymbol{W}^\ell \sim_{\text{i.i.d.}} \mathcal{N}(0, 2/n)$ if $\ell \in [L]$ and $\boldsymbol{W}^{L+1} \sim_{\text{i.i.d.}} \mathcal{N}(0, 1)$, independently of the samples $\boldsymbol{x}_1, \ldots, \boldsymbol{x}_N$, we consider the sequence of iterates

$$\boldsymbol{\theta}_{k+1}^N = \boldsymbol{\theta}_k^N - \tau \widetilde{\nabla} \mathcal{L}_{\mu^N}(\boldsymbol{\theta}_k^N), \tag{A.1}$$

where $\tau > 0$ is a step size, and $\widetilde{\nabla} \mathcal{L}_{\mu^N}$ represents a 'formal gradient' of the loss $\mathcal{L}_{\mu^N}$, which we *define* as follows: first, we define formal gradients of the network output by

$$\widetilde{\nabla}_{\boldsymbol{W}^\ell} f_{\boldsymbol{\theta}}(\boldsymbol{x}) = \boldsymbol{\beta}_{\boldsymbol{\theta}}^{\ell-1}(\boldsymbol{x}) \boldsymbol{\alpha}_{\boldsymbol{\theta}}^{\ell-1}(\boldsymbol{x})^*$$

for $\ell \in [L]$ and $\boldsymbol{x} \in \mathcal{M}$, where we have introduced the definitions

$$\boldsymbol{\beta}_{\boldsymbol{\theta}}^\ell(\boldsymbol{x}) = \left( \boldsymbol{W}^{L+1} \boldsymbol{P}_{I_L(\boldsymbol{x})} \boldsymbol{W}^L \boldsymbol{P}_{I_{L-1}(\boldsymbol{x})} \ldots \boldsymbol{W}^{\ell+2} \boldsymbol{P}_{I_{\ell+1}(\boldsymbol{x})} \right)^*$$

for $\ell = 0, 1, \ldots, L - 1$, and where we additionally define

$$I_\ell(\boldsymbol{x}) = \text{supp}\left( \mathbb{1}_{\boldsymbol{\alpha}_{\boldsymbol{\theta}}^\ell(\boldsymbol{x}) > 0} \right), \qquad \boldsymbol{P}_{I_\ell(\boldsymbol{x})} = \sum_{i \in I_\ell(\boldsymbol{x})} \boldsymbol{e}_i \boldsymbol{e}_i^*$$

for the orthogonal projection onto the set of coordinates where the $\ell$-th activation at input $\boldsymbol{x}$ is positive. We call the vectors $\boldsymbol{\beta}_{\boldsymbol{\theta}}^\ell(\boldsymbol{x})$ the *backward features* or *backward activations*—they correspond to the backward pass of our neural network. We also define

$$\widetilde{\nabla}_{\boldsymbol{W}^{L+1}} f_{\boldsymbol{\theta}}(\boldsymbol{x}) = \boldsymbol{\alpha}_{\boldsymbol{\theta}}^L(\boldsymbol{x})^*.$$

We then define the formal gradient of the loss $\mathcal{L}_{\mu^N}$ by

$$\widetilde{\nabla} \mathcal{L}_{\mu^N}(\boldsymbol{\theta}) = \int_{\mathcal{M}} \widetilde{\nabla} f_{\boldsymbol{\theta}}(\boldsymbol{x}) \zeta_{\boldsymbol{\theta}}(\boldsymbol{x}) \, d\mu^N(\boldsymbol{x}).$$

Let us emphasize again that the expressions above are definitions, not gradients in the analytical sense: we introduce these definitions to cope with nonsmoothness of the ReLU $[\,\cdot\,]_+$. On the other hand, our formal gradient definitions coincide with the expressions one obtains by applying the chain rule to differentiate $\mathcal{L}_{\mu^N}$ at points where the ReLU is differentiable, and we will make use of this fact to proceed with these formal gradients in a manner almost identical to the differentiable setting.

We reiterate here our notational conventions for quantities evaluated at these iterates: we denote evaluation of quantities such as the features and prediction error at parameters along the gradient descent trajectory using a subscript $k$, with an omitted subscript denoting evaluation at the initial $k = 0$ parameters, and we add a superscript $N$ to parameters such as the prediction error to emphasize that they are evaluated at the parameters generated by (A.1). For example, in this notation we express $\zeta_{\boldsymbol{\theta}_k^N}$ as $\zeta_k^N$. In addition, we use $\boldsymbol{\theta}_0$ to denote the initial parameters $\boldsymbol{\theta}_0^N$. We emphasize the dependence of certain quantities on these random initial parameters notationally, including the initial network function $f_{\boldsymbol{\theta}_0}$.

### A.2   REGARDING THE DATA MANIFOLDS

We now provide additional details regarding our assumptions on the data manifolds. For background on curves and more broadly Riemannian manifolds, we refer the reader to (Lee, 2018; Absil et al., 2009). We assume that $\mathcal{M} = \mathcal{M}_+ \cup \mathcal{M}_-$, where $\mathcal{M}_+$ and $\mathcal{M}_-$ are two disjoint complete connected[5] Riemannian submanifolds of the unit sphere $\mathbb{S}^{n_0-1}$, with $n_0 \geq 3$. In particular, $\mathcal{M}_\pm$ are compact. We take as metric on these manifolds the metric induced by that of the sphere, which we take in turn as that induced by the euclidean metric on $\mathbb{R}^{n_0}$. We write $\mu_+^\infty$ and $\mu_-^\infty$ for the measures on $\mathcal{M}_+$

---

[5]Certain parts of our argument, such as the concentration result Theorem B.2, are naturally applicable to cases where $\mathcal{M}_\pm$ themselves have a finite number of connected components with a mild dependence on this number, and we state them as such. We skip this extra generality in our dynamics arguments to avoid an additional 'juggling act' that would obscure the main ideas.

and $\mathcal{M}_-$ (respectively) induced by the data measure $\mu^\infty$, and we assume that $\mu^\infty$ admits a density $\rho$ with respect to the Riemannian measure on $\mathcal{M}$, writing $\rho_+$ and $\rho_-$ for the densities on $\mathcal{M}_\pm$ induced by the density $\rho$. When $d_0 = 1$, we add additional structural assumptions to the above: we assume that $\mathcal{M}_\pm$ are smooth, simple, regular curves.

Concretely, that $\mathcal{M}$ admits a density $\rho$ with respect to the Riemannian measure means that

$$1 = \int_{\mathcal{M}} \mathrm{d}\mu^\infty(\boldsymbol{x}) = \int_{\mathcal{M}_+} \rho_+(\boldsymbol{x})\, \mathrm{d}V_+(\boldsymbol{x}) + \int_{\mathcal{M}_-} \rho_-(\boldsymbol{x})\, \mathrm{d}V_-(\boldsymbol{x}).$$

When $d_0 = 1$, because $\mathcal{M}_\pm$ are smooth regular curves, they admit global unit-speed parameterizations with respect to arc length $\boldsymbol{\gamma}_\pm : I_\pm \to \mathbb{S}^{n_0-1}$, where $I_\pm$ are intervals of the form $[0, \mathrm{len}(\mathcal{M}_\pm)]$. In this setting, the curvature constraint is expressed as

$$\max\left\{ \sup_{s\in I_+} \left\| \boldsymbol{\gamma}_+''(s) \right\|_2, \sup_{s\in I_-} \left\| \boldsymbol{\gamma}_-''(s) \right\|_2 \right\} \leq \kappa,$$

and we observe that the fact that $\mathcal{M}_\pm$ are sphere curves implies $\kappa \geq 1$.[6] Exploiting the coordinate representation of the Riemannian measure and the fixed inherited metric from $\mathbb{R}^{n_0}$, we thus have

$$\int_{\mathcal{M}_\pm} \rho_\pm(\boldsymbol{x})\, \mathrm{d}V_\pm(\boldsymbol{x}) = \int_{I_\pm} \rho_\pm \circ \gamma_\pm(t) \|\gamma_\pm'(t)\|_2\, \mathrm{d}t = \int_{I_\pm} \rho_\pm \circ \gamma_\pm(t)\, \mathrm{d}t.$$

We will exploit this formula in the sequel to compare between $L_\mu^p(\mathcal{M})$ and $L^p(\mathcal{M})$ norms of functions defined on the manifold. More generally, we will frequently make use of similar reasoning that leverages the existence of unit-speed parameterizations for the curves.

For clarity we rewrite the global regularity condition: we assume there exist constants $0 < c_\lambda \leq 1$, $K_\lambda \geq 1$ such that

$$\forall s \in (0, c_\lambda/\kappa], \, (\boldsymbol{x}, \boldsymbol{x}') \in \mathcal{M}_\star \times \mathcal{M}_\star, \star \in \{+, -\} \quad : \quad \angle(\boldsymbol{x}, \boldsymbol{x}') \leq s \Rightarrow \mathrm{dist}_{\mathcal{M}}(\boldsymbol{x}, \boldsymbol{x}') \leq K_\lambda s, \tag{A.2}$$

where $\mathrm{dist}_{\mathcal{M}}$ denotes the Riemannian distance between points in a common connected component, and we define $C_\lambda = K_\lambda^2/c_\lambda^2$. Because $\mathcal{M}_\pm$ are simple curves, they do not self-intersect; the assumption (A.2) gives a quantitative characterization of how far the curves are from self-intersecting. We illustrate how the associated constants can be obtained from the assumption that the manifolds are simple curves: for either $\star \in \{+, -\}$, consider a connected component $\mathcal{M}_\star \subset \mathcal{M}$, and for any $0 < s \leq \mathrm{len}(\mathcal{M}_\star)$, define

$$r_\star(s) = \inf_{\substack{\boldsymbol{x}, \boldsymbol{x}' \in \mathcal{M}_\star \times \mathcal{M}_\star, \\ \mathrm{dist}_{\mathcal{M}}(\boldsymbol{x}, \boldsymbol{x}') > s}} \angle(\boldsymbol{x}, \boldsymbol{x}').$$

If $r_\star(s) = 0$, by compactness we can construct a sequence of pairs of points that converges to $r_\star(s)$, but this would imply that $\mathcal{M}_\star$ is self-intersecting, contradicting our assumption that it is simple. It follows that $r_\star(s) > 0$ for any value of $s$. If we now define $\tilde{K}_s = r_\star(s)/s$, it follows that for any $(\boldsymbol{x}, \boldsymbol{x}') \in \mathcal{M}_\star \times \mathcal{M}_\star$,

$$\angle(\boldsymbol{x}, \boldsymbol{x}') \leq s \Rightarrow \mathrm{dist}_{\mathcal{M}}(\boldsymbol{x}, \boldsymbol{x}') \leq \tilde{K}_s s.$$

Our regularity assumption implies that a single such constant holds for a range of scales below the curvature scale, which is a mild assumption since $\tilde{K}_s$ approaches 1 as $s$ approaches 0.

### A.3  REGARDING THE INITIALIZATION

The manner in which we have defined our initial random neural network $f_{\boldsymbol{\theta}_0}$ is sometimes referred to as "fan-out initialization" in the literature—it guarantees that feature norms are preserved from layer to layer in the network, and thereby avoids the vanishing and exploding gradient problems. The difference between this initialization and the so called "standard" or "fan-in" initialization is only in the first and last layer weights, yet in a sufficiently deep network trained in the NTK regime the effect of any single layer is negligible and the dynamics of our network will be essentially identical to one with standard initialization. On the other hand, following the work of Jacot et al.

---

[6]We point out that the curvature of the manifolds does not enter into the proof of the concentration result Theorem B.2, so there is no ambiguity in discussing curvature only in the context of curves.

(2018), it has become common in the theoretical literature to consider a different construction of the neural network called "NTK parameterization", which is in some ways more convenient for theoretical analysis. In particular, Arora et al. (2019b) prove their results on NTK concentration using this parameterization; to facilitate a comparison between our concentration result (Theorem 2) and theirs, we discuss the connection between fan-out and NTK parameterization in this section. This material is well-known and no doubt can be found already in the literature, but we believe it may be helpful to translate it into our notation.

Recall our definitions for the weights and features in our neural network: we have $\boldsymbol{W}^\ell \sim_{\text{i.i.d.}} \mathcal{N}(0, 2/n)$ if $\ell \in \{0, 1 \ldots, L\}$ and $\boldsymbol{W}^{L+1} \sim_{\text{i.i.d.}} \mathcal{N}(0, 1)$, with features defined for $\ell = 0, 1, \ldots, L$ by

$$\boldsymbol{\alpha}^\ell(\boldsymbol{x}) = \begin{cases} \boldsymbol{x} & \ell = 0 \\ \left[ \boldsymbol{W}^\ell \boldsymbol{\alpha}^{\ell-1}(\boldsymbol{x}) \right]_+ & \text{otherwise,} \end{cases}$$

and output $f_{\boldsymbol{\theta}_0}(\boldsymbol{x}) = \boldsymbol{W}^{L+1} \boldsymbol{\alpha}^L(\boldsymbol{x})$. Within this section—and only within this section—we shall define auxiliary weights by $\boldsymbol{G}^{(1)} \in \mathbb{R}^{n \times n_0}$, $\boldsymbol{G}^{(\ell)} \in \mathbb{R}^{n \times n}$ for integer $1 < \ell < L + 1$, and $\boldsymbol{G}^{(L+1)} \in \mathbb{R}^{1 \times n}$, with distributions $\boldsymbol{G}^{(\ell)} \sim_{\text{i.i.d.}} \mathcal{N}(0, 1)$ for $\ell \in \{0, 1 \ldots, L + 1\}$, independent of everything else in the problem. As before, for $\ell \in \{0, 1, \ldots, L\}$ we use $\boldsymbol{\alpha}_{\text{NTK}}^{(\ell)}(\boldsymbol{x})$ to denote the layer-$\ell$ features:

$$\boldsymbol{\alpha}_{\text{NTK}}^{(\ell)}(\boldsymbol{x}) = \begin{cases} \boldsymbol{x} & \ell = 0 \\ \left[ \boldsymbol{G}^{(\ell)} \boldsymbol{\alpha}_{\text{NTK}}^{(\ell-1)}(\boldsymbol{x}) \right]_+ & \text{otherwise.} \end{cases}$$

This network's output will be written

$$f_{\text{NTK}}(\boldsymbol{x}) = \left( \prod_{\ell=1}^L \sqrt{\frac{2}{n}} \right) \boldsymbol{G}^{(L+1)} \boldsymbol{\alpha}_{\text{NTK}}^{(L)}(\boldsymbol{x}).$$

By 1-homogeneity (absolute) of $\sigma$, it follows that $f_{\boldsymbol{\theta}_0} \overset{d}{=} f_{\text{NTK}}$. As the notation suggests, the network $f_{\text{NTK}}$ corresponds to a "NTK parameterization" network—although this network and $f_{\boldsymbol{\theta}_0}$ are equivalent in terms of predictions, their "gradients" are not equivalent. The NTK for the NTK parameterization network is obtained by differentiating (at points of differentiability): after calculating (as in Lemma B.8), we introduce notation as we did for the fan-out parameterization network in Appendix A.1, so that

$$\Theta_{\text{NTK}}(\boldsymbol{x}, \boldsymbol{x}') = \left\langle \widetilde{\nabla} f_{\text{NTK}}(\boldsymbol{x}), \widetilde{\nabla} f_{\text{NTK}}(\boldsymbol{x}') \right\rangle,$$

with (for $\ell = 1, \ldots, L + 1$)

$$\widetilde{\nabla}_{\boldsymbol{G}^{(\ell)}} f_{\text{NTK}}(\boldsymbol{x}) = \left( \prod_{\ell=1}^L \sqrt{\frac{2}{n}} \right) \boldsymbol{\beta}_{\text{NTK}}^{(\ell-1)}(\boldsymbol{x}) \boldsymbol{\alpha}_{\text{NTK}}^{(\ell-1)}(\boldsymbol{x})^*$$

where

$$\boldsymbol{\beta}_{\text{NTK}}^{(\ell)}(\boldsymbol{x}) = \begin{cases} \left( \boldsymbol{G}^{(L+1)} \boldsymbol{P}_{I_L^{\text{NTK}}(\boldsymbol{x})} \boldsymbol{G}^{(L)} \boldsymbol{P}_{I_{L-1}^{\text{NTK}}(\boldsymbol{x})} \cdots \boldsymbol{G}^{(\ell+2)} \boldsymbol{P}_{I_{\ell+1}^{\text{NTK}}(\boldsymbol{x})} \right)^* & \ell = 0, 1, \ldots, L - 1 \\ 1 & \ell = L, \end{cases}$$

and

$$I_\ell^{\text{NTK}}(\boldsymbol{x}) = \text{supp}\left( \mathbb{1}_{\boldsymbol{\alpha}_{\text{NTK}}^{(\ell)}(\boldsymbol{x}) > 0} \right).$$

We shall relate the NTK parameterization NTK $\Theta_{\text{NTK}}$ to our fan-out parameterization NTK $\Theta$ using homogeneity of the ReLU. First, let us observe that

$$\left\{ i \in [n] \mid \left( \boldsymbol{\alpha}^\ell(\boldsymbol{x}) \right)_i > 0 \right\} \overset{d}{=} \left\{ i \in [n] \mid \left( \boldsymbol{\alpha}_{\text{NTK}}^{(\ell)}(\boldsymbol{x}) \right)_i > 0 \right\}.$$

because $[\cdot]_+$ is 1-homogeneous and we have $\boldsymbol{G}^{(\ell)} \overset{d}{=} \sqrt{n/2} \boldsymbol{W}^\ell$ when $\ell \leq L$. For $\ell \in \{0, 1, \ldots, L\}$, we note that both $\boldsymbol{\alpha}^\ell(\boldsymbol{x})$ and $\boldsymbol{\alpha}_{\text{NTK}}^{(\ell)}(\boldsymbol{x})$ depend only on the parameters $(\boldsymbol{W}^1, \ldots, \boldsymbol{W}^\ell)$ and $(\boldsymbol{G}^{(1)}, \ldots, \boldsymbol{G}^{(\ell)})$, respectively. If we write $\boldsymbol{\theta} = (\boldsymbol{W}^1, \ldots, \boldsymbol{W}^L)$ and $\boldsymbol{\theta}_{\text{NTK}} = (\boldsymbol{G}^{(1)}, \ldots, \boldsymbol{G}^{(L)})$, then it follows that $\boldsymbol{\alpha}^\ell(\boldsymbol{x})$ is a $\ell$-homogeneous function of $\boldsymbol{\theta}$ (and likewise for $\boldsymbol{\alpha}_{\text{NTK}}^{(\ell)}(\boldsymbol{x})$). In

addition, the projection matrices $\boldsymbol{P}_{I_\ell}(\boldsymbol{x})$ are 0-homogeneous functions of $\boldsymbol{\theta}$, and so taking $\ell \in \{0, 1, \dots, L-1\}$ and counting parameters in the definitions of $\boldsymbol{\beta}^\ell(\boldsymbol{x})$ and $\boldsymbol{\beta}_{\mathrm{NTK}}^{(\ell)}(\boldsymbol{x})$ implies that these two functions are $(L - \ell - 1)$-homogeneous functions of $\boldsymbol{\theta}$ and $\boldsymbol{\theta}_{\mathrm{NTK}}$, respectively. Of course, for $\ell = L$, they are 0-homogeneous. Thus, using that $\boldsymbol{G}^{(\ell)} \overset{d}{=} \sqrt{n/2}\boldsymbol{W}^\ell$ for $\ell \le L$ again, we obtain

$$\widetilde{\nabla}_{\boldsymbol{G}^{(\ell)}} f_{\mathrm{NTK}}(\boldsymbol{x}) \overset{d}{=} \begin{cases} \sqrt{2/n}\widetilde{\nabla}_{\boldsymbol{W}^\ell} f_{\boldsymbol{\theta}_0}(\boldsymbol{x}) & \ell = 0, 1, \dots, L \\ \widetilde{\nabla}_{\boldsymbol{W}^\ell} f_{\boldsymbol{\theta}_0}(\boldsymbol{x}) & \ell = L+1. \end{cases}$$

Although we have argued equidistributionality above for each index $\ell$ separately for simplicity, the elementary nature of our arguments (we are just moving scalars around) and the statistical dependencies across gradients allows us to apply the same argument 'in parallel' to the sum of inner products between gradients, yielding

$$\Theta_{\mathrm{NTK}}(\boldsymbol{x}, \boldsymbol{x}') \overset{d}{=} \langle \boldsymbol{\alpha}^L(\boldsymbol{x}), \boldsymbol{\alpha}^L(\boldsymbol{x}') \rangle + \frac{2}{n} \sum_{\ell=1}^L \langle \boldsymbol{\alpha}^{\ell-1}(\boldsymbol{x}), \boldsymbol{\alpha}^{\ell-1}(\boldsymbol{x}') \rangle \langle \boldsymbol{\beta}^{\ell-1}(\boldsymbol{x}), \boldsymbol{\beta}^{\ell-1}(\boldsymbol{x}') \rangle.$$

This expression makes it immediately clear that our concentration framework proves sharp concentration of the NTK of a uniform-width NTK parameterization feedforward ReLU network that improves over the results of Arora et al. (2019b) when the data are on the sphere[7] —a simple adaptation of the proof of Theorem B.3 will suffice.

## A.4  PROOF OUTLINE FOR THEOREM 1

In Appendix B, we prove a slightly more general version of Theorem 1 in Theorem B.1. Here, we give a brief outline of the proof of this result.

Proving the separation property essentially requires us to obtain control of $\|\zeta_k^N\|_{L^\infty(\mathcal{M})}$, and by an interpolation inequality (Lemma B.14) it suffices to control the generalization error $\|\zeta_k^N\|_{L_{\mu^\infty}^2}$ and the smoothness (measured through the Lipschitz constant) of $\zeta_k^N$. We start with the generalization error, picking up from where we left off at the end of Section 2.2: the triangle inequality gives

$$\left\| \zeta_k^N \right\|_{L_{\mu^\infty}^2} \le \left\| \zeta_k^\infty \right\|_{L_{\mu^\infty}^2} + \left\| \zeta_k^\infty - \zeta_k^N \right\|_{L_{\mu^\infty}^2}, \tag{A.3}$$

which allows us to divide the analysis into two subproblems: characterizing the nominal dynamics (Lemmas B.6 and B.12), and the nominal-to-finite transition (Lemma B.7). Beginning with the nominal dynamics, we use (2.2) to write

$$\zeta_k^\infty = (\mathrm{Id} - \tau \boldsymbol{\Theta})^k [\zeta],$$

where $\boldsymbol{\Theta}$ denotes the operator on $L_{\mu^\infty}^2$ corresponding to integration against the kernel $\Theta$ and $\mathrm{Id}$ denotes the identity operator. The definition of $\Theta$ and compactness of $\mathcal{M}$ imply that $\boldsymbol{\Theta}$ is a positive, compact operator (Lemma B.9), so these dynamics are stable when $\tau$ is chosen larger than the operator norm of $\boldsymbol{\Theta}$. However, the rate of decrease of $\|\zeta_k^\infty\|_{L_{\mu^\infty}^2}$ with $k$ could still be extremely slow if the initial error $\zeta$ has significant components in the direction of eigenfunctions of $\boldsymbol{\Theta}$ corresponding to small eigenvalues, and because $\boldsymbol{\Theta}$ acts roughly like a convolution operator, we expect there to exist eigenvalues arbitrarily close to zero. By solving the certificate problem (2.3), we can assert that misalignment does not occur. To solve the certificate problem, as we describe in Section 3.2, we work with analytically-convenient approximations for $\boldsymbol{\Theta}$ and $\zeta$: the exact definitions of these approximations $\hat{\boldsymbol{\Theta}}$ and $\hat{\zeta}$ are given in Appendix A.5.2, and we prove their suitability as approximations in Theorem B.2 (a slightly more general version of Theorem 2) and Lemma D.11, respectively. As we have discussed in Section 3.1, our rates of concentration for $\boldsymbol{\Theta}$ about $\hat{\boldsymbol{\Theta}}$ are essentially optimal—the poor rates that end up appearing in Theorem B.1 are set by later parts of the argument.

With our approximation to $\boldsymbol{\Theta}$ justified, we show that for any sufficiently small step size $\tau$ and number of iterations $k$, solving the certificate problem (2.3) guarantees appropriate decrease of the

---

[7]The results of Arora et al. (2019b) apply to data of norm no larger than 1, but it is straightforward to extend our results for spherical data to this setting, using the 1-homogeneity of $\Theta$ in each argument (as a kernel on the entire ambient space $\mathbb{R}^{n_0} \times \mathbb{R}^{n_0}$) to write $\Theta(\boldsymbol{x}, \boldsymbol{x}') = \|\boldsymbol{x}\|_2 \|\boldsymbol{x}'\|_2 \Theta(\boldsymbol{x}/\|\boldsymbol{x}\|_2, \boldsymbol{x}'/\|\boldsymbol{x}'\|_2)$.

nominal generalization error; additional details are discussed in Section 3.2. The key property that we use in constructing certificates in Proposition B.4 (the 'appendix version' of Proposition 1) is that as the depth $L$ increases, the kernel $\hat{\Theta}$ sharpens and localizes (Fig. 2a): the conditions on $L$ in Theorem B.1 guarantee that the sharpness is sufficient to ensure that the cross-manifold integrals in the certificate problem are small in magnitude, which leads to rapid decrease of the nominal error. Our precise characterization of this phenomenon is presented in Appendix C.

To complete the proof, we will justify the nominal-to-finite transition in (A.3). Starting from the update equations (2.1) and (2.2), subtracting and rearranging gives an update equation for the difference:

$$\zeta_k^N - \zeta_k^\infty = (\mathrm{Id} - \tau\boldsymbol{\Theta}) \left[\zeta_{k-1}^N - \zeta_{k-1}^\infty\right] - \tau\boldsymbol{\Theta}_{k-1}^N \left[\zeta_{k-1}^N\right] + \tau\boldsymbol{\Theta} \left[\zeta_{k-1}^N\right].$$

In particular, if $\tau$ is chosen less than the operator norm of $\boldsymbol{\Theta}$, we can take norms on both sides of the previous equation, apply the triangle inequality, then exploit a telescoping series cancellation to obtain the difference bound

$$\left\|\zeta_k^\infty - \zeta_k^N\right\|_{L_{\mu^\infty}^2} \leq \tau \sum_{s=0}^{k-1} \left\|\int_{\mathcal{M}} \Theta_s^N(\,\cdot\,,\boldsymbol{x}')\zeta_s^N(\boldsymbol{x}')\,\mathrm{d}\mu^N(\boldsymbol{x}') - \int_{\mathcal{M}} \Theta(\,\cdot\,,\boldsymbol{x}')\zeta_s^N(\boldsymbol{x}')\,\mathrm{d}\mu^\infty(\boldsymbol{x}')\right\|_{L_{\mu^\infty}^2}.$$
$$\text{(A.4)}$$

There are two obstacles to controlling the norm terms on the RHS of (A.4): the kernels $\Theta_s^N$ are distinct from the kernel $\Theta$ due to changes in the weights that occur during training, and the empirical measure $\mu^N$ incurs a sampling error relative to the population measure $\mu^\infty$. To address the first challenge, we measure the changes to the NTK during training in a worst-case fashion as

$$\Delta_k^N = \max_{i \in \{0,1,\dots,k\}} \left\|\Theta_i^N - \Theta\right\|_{L^\infty(\mathcal{M} \times \mathcal{M})},$$

and train in the *NTK regime*, where the network width $n$ is larger than a large polynomial in the depth $L$ and the total training time $k\tau$ is no larger than $L/n$. These conditions imply that with high probability $\Delta_k^N$ is no larger than a constant multiple of $n^{1-\delta}\,\mathrm{poly}(L, d_0)$ for a small constant $\delta > 0$, so that the amortized changes during training $k\tau\Delta_k^N$ can be made small by sufficient overparameterization. We provide full details of this argument in Appendix F. By the preceding argument, we can use the triangle inequality and Jensen's inequality to pass from the norm term in (A.4) to a difference-of-measures term which integrates against $\Theta$, and by Theorem B.2, we can replace the integration against $\Theta$ by an integration against a smooth, deterministic kernel, which leads to a bound

$$\left\|\zeta_k^\infty - \zeta_k^N\right\|_{L_{\mu^\infty}^2} \leq R_k(n, L, d_0) + \tau \sum_{s=0}^{k-1} \left\|\int_{\mathcal{M}} \psi_1(\angle(\,\cdot\,,\boldsymbol{x}'))\zeta_s^N(\boldsymbol{x}') \left(\mathrm{d}\mu^N(\boldsymbol{x}') - \mathrm{d}\mu^\infty(\boldsymbol{x}')\right)\right\|_{L_{\mu^\infty}^2},$$

where $R_k$ is a residual term that we argue is small in the NTK regime with high probability, and for concision we write $\psi_1$ to denote the function of $\angle(\boldsymbol{x}, \boldsymbol{x}')$ that appears in Theorem B.2. To control the remaining term, we make use of a basic result from optimal transport theory, which states that for any probability measure $\mu$ on the Borel sets of a metric space $X$ and corresponding empirical measure $\mu^N$, one has for every Lipschitz function $f$

$$\int_X f(x) \left(\mathrm{d}\mu(x) - \mathrm{d}\mu^N(x)\right) \leq \|f\|_{\mathrm{Lip}} \mathcal{W}\left(\mu, \mu^N\right),$$

where $\mathcal{W}(\,\cdot\,,\,\cdot\,)$ denotes the 1-Wasserstein metric, and concentration inequalities for empirical measures in the 1-Wasserstein metric (Weed & Bach, 2019). To apply this result to our setting, it is necessary to control the change throughout training of the Lipschitz constant of $\zeta_k^N$, and one must also account for the fact that the metric space in our setting is $\mathcal{M}$, which has two distinct connected components. We treat the first issue using an inductive argument, and our treatment of the second issue (Lemma B.13) leads to the dependence on the degree of class imbalance demonstrated in the constant $C_{\mu^\infty}$ in Theorem B.1.

## A.5 NOTATION

### A.5.1 GENERAL NOTATION

If $n \in \mathbb{N}$, we write $[n] = \{1, \dots, n\}$. We generally use bold notation $\boldsymbol{x}$, $\boldsymbol{A}$ for vectors, matrices, and operators and non-bold notation for scalars and scalar-valued functions. For a vector $\boldsymbol{x}$ or a matrix

$\boldsymbol{A}$, we will write entries as either $x_j$ or $A_{ij}$, or $(\boldsymbol{x})_j$ or $(\boldsymbol{A})_{ij}$; we will occasionally index the rows or columns of $\boldsymbol{A}$ similarly as $(\boldsymbol{A})_i$ or $(\boldsymbol{A})_j$, with the particular meaning made clear from context. We write $[x]_+ = \max\{x, 0\}$ for the ReLU activation function; if $\boldsymbol{x}$ is a vector, we write $[\boldsymbol{x}]_+$ to denote the vector given by the application of $[\,\cdot\,]_+$ to each coordinate of $\boldsymbol{x}$, and we will generally adopt this convention for applying scalar functions to vectors. If $\boldsymbol{x}, \boldsymbol{x}' \in \mathbb{R}^n$ are nonzero, we write $\angle(\boldsymbol{x}, \boldsymbol{x}') = \cos^{-1}(\langle \boldsymbol{x}, \boldsymbol{x}' \rangle / \|\boldsymbol{x}\|_2 \|\boldsymbol{x}'\|_2)$ for the angle between $\boldsymbol{x}$ and $\boldsymbol{x}'$.

The vectors $(\boldsymbol{e}_i)$ denote the canonical basis for $\mathbb{R}^n$. We write $\langle \boldsymbol{x}, \boldsymbol{y} \rangle = \sum_i x_i y_i$ for the euclidean inner product on $\mathbb{R}^n$, and if $0 < p < +\infty$ we write $\|\boldsymbol{x}\|_p = (\sum_i |x_i|^p)^{1/p}$ for the $\ell^p$ norms (when $p \geq 1$) on $\mathbb{R}^n$. We also write $\|\boldsymbol{x}\|_0 = |\{i \in [n] \mid x_i \neq 0\}|$ and $\|\boldsymbol{x}\|_\infty = \max_{i \in [n]} |x_i|$. The unit ball in $\mathbb{R}^n$ is written $\mathbb{B}^n = \{\boldsymbol{x} \in \mathbb{R}^n \mid \|\boldsymbol{x}\|_2 \leq 1\}$, and we denote its (topological) boundary, the unit sphere, as $\mathbb{S}^{n-1}$. We reserve the notation $\|\cdot\|$ for the operator norm of a $m \times n$ matrix $\boldsymbol{A}$, defined as $\|\boldsymbol{A}\| = \sup_{\|\boldsymbol{x}\|_2 \leq 1} \|\boldsymbol{A}\boldsymbol{x}\|_2$; more generally, we write $\|\boldsymbol{A}\|_{\ell^p \to \ell^q} = \sup_{\|\boldsymbol{x}\|_p \leq 1} \|\boldsymbol{A}\boldsymbol{x}\|_q$ for the corresponding induced matrix norm. For $m \times n$ matrices $\boldsymbol{A}$ and $\boldsymbol{B}$, we write $\langle \boldsymbol{A}, \boldsymbol{B} \rangle = \mathrm{tr}(\boldsymbol{A}^* \boldsymbol{B})$ for the standard inner product, where $\boldsymbol{A}^*$ denotes the transpose of $\boldsymbol{A}$, and $\|\boldsymbol{A}\|_F = \sqrt{\langle \boldsymbol{A}, \boldsymbol{A} \rangle}$ for the Frobenius norm of $\boldsymbol{A}$.

The Banach space of (equivalence classes of) real-valued measurable functions on a measure space $(X, \mu)$ satisfying $(\int_X |f|^p \, d\mu)^{1/p} < +\infty$ is written $L_\mu^p(X)$ or simply $L^p$ if the space and/or measure is clear from context; we write $\|\cdot\|_{L^p}$ for the associated norm, and $\langle \cdot, \cdot \rangle_{L^2}$ for the associated inner product when $p = 2$, with the adjoint operation denoted by $^*$. For an operator $\mathcal{T} : L_\mu^p \to L_\nu^q$, we write $\mathcal{T}[f]$ to denote the image of $f$ under $\mathcal{T}$, $\mathcal{T}^i$ to denote the operator that applies $\mathcal{T}$ $i$ times, and $\|\mathcal{T}\|_{L_\mu^p \to L_\nu^q} = \sup_{\|f\|_{L_\mu^p} \leq 1} \|\mathcal{T}[f]\|_{L_\nu^q}$. We use $\mathrm{Id}$ to denote the identity operator, i.e. $\mathrm{Id}[g] = g$ for every $g \in L_\mu^p$. We say that $\mathcal{T}$ is positive if $\langle f, \mathcal{T}[f] \rangle_{L^2} \geq 0$ for all $f \in L^2$; for example, the identity operator is positive.

For an event $\mathcal{E}$ in a probability space, we write $\mathbb{1}_\mathcal{E}$ to denote the indicator random variable that takes the value 1 if $\omega \in \mathcal{E}$ and 0 otherwise. If $\sigma > 0$, by $\boldsymbol{g} \sim \mathcal{N}(\boldsymbol{0}, \sigma^2 \boldsymbol{I})$ we mean that $\boldsymbol{g} \in \mathbb{R}^n$ is distributed according to the standard i.i.d. gaussian law with variance $\sigma^2$, i.e., it admits the density $(2\pi\sigma^2)^{-n/2} \exp(-\|\boldsymbol{x}\|_2^2 / (2\sigma^2))$ with respect to Lebesgue measure on $\mathbb{R}^n$; we will occasionally write this equivalently as $\boldsymbol{g} \sim_{\text{i.i.d.}} \mathcal{N}(0, \sigma^2)$. We use $\overset{d}{=}$ to denote the "identically-distributed" equivalence relation.

We use "numerical constant" and "absolute constant" interchangeably for numbers that are independent of all problem parameters. Throughout the text, unless specified otherwise we use $c, c', c''$, $C, C', C'', K, K', K''$, and so on to refer to numerical constants whose value may change from line to line within a proof. Numerical constants with numbered subscripts $C_1, C_2, \ldots$ and so on will have values fixed at the scope of the proof of a single result, unless otherwise specified. We generally use lower-case letters to refer to numerical constants whose value should be small, and upper case for those that should be large; we will generally use $K$, $K'$ and so on to denote numerical constants involved in lower bounds on the size of parameters required for results to be valid. If $f$ and $g$ are two functions, the notation $f \lesssim g$ means that there exists a numerical constant $C > 0$ such that $f \leq Cg$; the notation $f \gtrsim g$ means that there exists a numerical constant $C > 0$ such that $f \geq Cg$; and when both are true simultaneously we write $f \asymp g$. If $f$ is a real-valued function with sufficient differentiability properties, we will write both $f'$ and $\dot{f}$ for the derivative of $f$, and when higher derivatives are available we will occasionally denote them by $f^{(n)}$, with this usage specifically made clear in context. For a metric space $X$ and a Lipschitz function $f : X \to \mathbb{R}$, we write $\|f\|_{\mathrm{Lip}}$ to denote the minimal Lipschitz constant of $f$.

### A.5.2 SUMMARY OF OPERATOR AND ERROR DEFINITIONS

We collect some of the important definitions that appear throughout the main text and the appendices in this section. We begin with the NTK-type operators that appear in our analysis. Recall from Appendix A.1 our definition for the backward features: we have

$$\boldsymbol{\beta}_{\boldsymbol{\theta}}^\ell(\boldsymbol{x}) = \left( \boldsymbol{W}^{L+1} \boldsymbol{P}_{I_L(\boldsymbol{x})} \boldsymbol{W}^L \boldsymbol{P}_{I_{L-1}(\boldsymbol{x})} \cdots \boldsymbol{W}^{\ell+2} \boldsymbol{P}_{I_{\ell+1}(\boldsymbol{x})} \right)^*$$

for $\ell = 0, 1, \ldots, L - 1$, and where we additionally define

$$I_\ell(\boldsymbol{x}) = \mathrm{supp}\left( \mathbb{1}_{\boldsymbol{\alpha}_{\boldsymbol{\theta}}^\ell(\boldsymbol{x}) > 0} \right), \qquad \boldsymbol{P}_{I_\ell(\boldsymbol{x})} = \sum_{i \in I_\ell(\boldsymbol{x})} \boldsymbol{e}_i \boldsymbol{e}_i^*$$

for the orthogonal projection onto the set of coordinates where the $\ell$-th activation at input $\boldsymbol{x}$ is positive. "The" neural tangent kernel is defined as

$$\Theta(\boldsymbol{x}, \boldsymbol{x}') = \left\langle \widetilde{\nabla} f_{\boldsymbol{\theta}_0}(\boldsymbol{x}), \widetilde{\nabla} f_{\boldsymbol{\theta}_0}(\boldsymbol{x}') \right\rangle$$

$$= \left\langle \boldsymbol{\alpha}^L(\boldsymbol{x}), \boldsymbol{\alpha}^L(\boldsymbol{x}') \right\rangle + \sum_{\ell=0}^{L-1} \left\langle \boldsymbol{\alpha}^\ell(\boldsymbol{x}), \boldsymbol{\alpha}^\ell(\boldsymbol{x}') \right\rangle \left\langle \boldsymbol{\beta}^\ell(\boldsymbol{x}), \boldsymbol{\beta}^\ell(\boldsymbol{x}') \right\rangle,$$

with corresponding operator on $L^2_{\mu^\infty}(\mathcal{M})$

$$\boldsymbol{\Theta}[g](\boldsymbol{x}) = \int_{\mathcal{M}} \Theta(\boldsymbol{x}, \boldsymbol{x}') g(\boldsymbol{x}') \, \mathrm{d}\mu^\infty(\boldsymbol{x}').$$

As shown in Lemma B.8, this is *not* exactly the kernel that governs the dynamics of gradient descent: the relevant kernels in this context are defined as

$$\Theta_k^N(\boldsymbol{x}, \boldsymbol{x}') = \int_0^1 \left\langle \widetilde{\nabla} f_{\boldsymbol{\theta}_k^N}(\boldsymbol{x}'), \widetilde{\nabla} f_{\boldsymbol{\theta}_k^N - t\tau\widetilde{\nabla}\mathcal{L}_{\mu^N}(\boldsymbol{\theta}_k^N)}(\boldsymbol{x}) \right\rangle \mathrm{d}t.$$

We define operators $\boldsymbol{\Theta}_k^N$ on $L^2_{\mu^N}(\mathcal{M})$ corresponding to integration against these kernel in a manner analogous to the definition of $\boldsymbol{\Theta}$:

$$\boldsymbol{\Theta}_k^N[g](\boldsymbol{x}) = \int_{\mathcal{M}} \Theta_k^N(\boldsymbol{x}, \boldsymbol{x}') g(\boldsymbol{x}') \, \mathrm{d}\mu^N(\boldsymbol{x}').$$

We then move to the deterministic approximations for $\Theta$ that we develop: we define

$$\varphi(\nu) = \cos^{-1}((1 - \nu/\pi)\cos\nu + (1/\pi)\sin\nu),$$

which governs the angle evolution process in the initial random network, as studied in Appendix E, and write $\varphi^{(\ell)}$ to denote $\ell$-fold composition of $\varphi$ with itself. We define

$$\psi_1(\nu) = \frac{n}{2} \sum_{\ell=0}^{L-1} \cos\left(\varphi^{(\ell)}(\nu)\right) \prod_{\ell'=\ell}^{L-1} \left(1 - \frac{\varphi^{(\ell')}(\nu)}{\pi}\right),$$

which is the "output" of our main result on concentration, Theorem B.2, and

$$\psi(\nu) = \frac{n}{2} \sum_{\ell=0}^{L-1} \prod_{\ell'=\ell}^{L-1} \left(1 - \frac{\varphi^{(\ell')}(\nu)}{\pi}\right),$$

which is at the core of the certificate construction problem. We think of $\psi$ as an analytically-simpler version of $\psi_1$, with an approximation guarantee given in Lemma C.11. Throughout these appendices, we will make use of basic properties of $\psi_1$ and $\psi$ that follow from properties of $\varphi$ without explicit reference; the source material for these types of claims is Lemma E.5, which gives elementary properties of $\varphi$ (for example, that it takes values in $[0, \pi/2]$, which implies that $\psi$ and $\psi_1$ are no larger than $nL/2$). For derived estimates, we call the reader's attention to the contents of Appendix C.2.2; we will make explicit reference to these results when we need them, however. Although we have mentioned approximations $\hat{\Theta}$ and $\check{\Theta}$ in the main text, we will prefer in these appendices to explicitly reference $\psi$ and $\psi_1$ to avoid confusion; as an exception, we will use the $\check{\Theta}$ notation in Appendix C as discussed there. Our approximation for the initial prediction error is

$$\hat{\zeta}(\boldsymbol{x}) = -f_\star(\boldsymbol{x}) + \int_{\mathcal{M}} f_{\boldsymbol{\theta}_0}(\boldsymbol{x}') \, \mathrm{d}\mu^\infty(\boldsymbol{x}'), \tag{A.5}$$

where we recall $f_{\boldsymbol{\theta}_0}$ denotes the network function with the initial (random) weights. In particular, this approximates the network function with a constant, and the error as a piecewise constant function on $\mathcal{M}_\pm$. This approximation is justified in Lemma D.11.

# B    PROOFS OF THE MAIN RESULTS

## B.1    MAIN RESULTS

**Theorem B.1.** *Let $\mathcal{M}$ be a one-dimensional Riemannian manifold satisfying our regularity assumptions. For any $0 < \delta \leq 1/e$, choose $L$ so that*

$$L \geq C_1 \max\big\{ C_{\mu^\infty} \log^9(1/\delta) \log^{24}\left( C_{\mu^\infty} n_0 \log(1/\delta) \right), \kappa^2 C_\lambda \big\},$$

*let $N \geq L^{10}$, set $n = C_2 L^{99} \log^9(1/\delta) \log^{18}(L n_0)$, and fix $\tau > 0$ such that*

$$\frac{C_3}{nL^2} \leq \tau \leq \frac{C_4}{nL}.$$

*Then if there exists a function $g \in L^2_{\mu^\infty}(\mathcal{M})$ such that*

$$\|\Theta[g] - \zeta\|_{L^2_{\mu^\infty}(\mathcal{M})} \leq C_5 \frac{\sqrt{\log(1/\delta) \log(nn_0)}}{L \min\{\rho_{\min}^{q_{\text{cert}}}, \rho_{\min}^{-q_{\text{cert}}}\}}; \quad \|g\|_{L^2_{\mu^\infty}(\mathcal{M})} \leq C_6 \frac{\sqrt{\log(1/\delta) \log(nn_0)}}{n \rho_{\min}^{q_{\text{cert}}}},$$
(B.1)

*with probability at least $1 - \delta$ over the random initialization of the network and the i.i.d. sample from $\mu^\infty$, the parameters obtained at iteration $\lfloor L^{39/44}/(n\tau) \rfloor$ of gradient descent on the finite sample loss $\mathcal{L}_{\mu^N}$ yield a classifier that separates the two manifolds.*

*The constants $C_1, \dots, C_4 > 0$ depend only on the constants $q_{\text{cert}}, C_5, C_6 > 0$, the constants $\kappa$, $C_\lambda$ are respectively the extrinsic curvature constant and the global regularity constant defined in Section 2.1, and the constant $C_{\mu^\infty}$ is defined as $\max\{\rho_{\min}^q, \rho_{\min}^{-q}\}(1 + \rho_{\max})^6 \left( \min\{\mu^\infty(\mathcal{M}_+), \mu^\infty(\mathcal{M}_-)\} \right)^{-11/2}$, where $q = 11 + 8 q_{\text{cert}}$.*

*Proof.* The proof is an application of Lemma B.7, with suitable instantiations of the parameters of that result; to avoid clashing with the probability parameter $\delta$ in this theorem, we use $\varepsilon$ for the parameter $\delta$ appearing in Lemma B.7. Define $C_\rho = \max\{\rho_{\min}, \rho_{\min}^{-1}\}$. We will pick $q = 39/44$ and $\varepsilon = 5/47$, so that the relevant hypotheses of Lemma B.7 become (after worst-casing in the bound on $N$ somewhat for readability)

$$d \geq K \log(nn_0 C_{\mathcal{M}})$$

$$n \geq K' \max\left\{ L^{99} d^9 \log^9 L, \kappa^{2/5}, \left(\frac{\kappa}{c_\lambda}\right)^{1/3} \right\}$$

$$L \geq K'' \max\{ C_\rho^{2q_{\text{cert}}} d, \kappa^2 C_\lambda \}$$

$$N \geq K''' \frac{C_\rho^{133/18 + (152/27)q_{\text{cert}}}(1 + \rho_{\max})^{133/54}}{\min\{\mu^\infty(\mathcal{M}_+), \mu^\infty(\mathcal{M}_-)\}^{19/18}} d^{8/3} L^9 \log^3 L,$$

and the conclusion we will appeal to becomes

$$\mathbb{P}\left[ \left\| \zeta^N_{\lfloor L^{39/44}/(n\tau)\rfloor} \right\|_{L^\infty(\mathcal{M})} \leq \frac{CC_\rho^{1 + 2q_{\text{cert}}/3}(1 + \rho_{\max})^{1/2}}{\min\{\mu^\infty(\mathcal{M}_+), \mu^\infty(\mathcal{M}_-)\}^{1/2}} \frac{d^{3/4} \log^{4/3} L}{L^{1/11}} \right] \geq 1 - \frac{C'L e^{-cd}}{n\tau}.$$

Under our choice of $\tau$ and enforcing

$$L \geq \frac{(2C)^{11} C_\rho^{11 + 22 q_{\text{cert}}/3}(1 + \rho_{\max})^{11/2} d^{33/4} \log^{44/3} L}{(\min\{\mu^\infty(\mathcal{M}_+), \mu^\infty(\mathcal{M}_-)\})^{11/2}},$$
(B.2)

we have the equivalent result

$$\mathbb{P}\left[ \left\| \zeta^N_{\lfloor L^{39/44}/(n\tau)\rfloor} \right\|_{L^\infty(\mathcal{M})} \leq \frac{1}{2} \right] \geq 1 - L^3 e^{-cd}$$

$$\geq 1 - e^{-c'd},$$

where the last bound holds when $d \geq K \log L$, which is redundant with the hypotheses on $n$ and $d$ required to use Lemma B.7. Thus, when in addition $d \geq (1/c') \log(1/\delta)$, we obtain

$$\mathbb{P}\left[ \left\| \zeta^N_{\lfloor L^{39/44}/(n\tau)\rfloor} \right\|_{L^\infty(\mathcal{M})} \leq \frac{1}{2} \right] \geq 1 - \delta.$$
(B.3)

Therefore to conclude, we need only argue that our choices of $n$, $N$, $L$, $d$, and $\delta$ in the theorem statement suffice to satisfy the hypotheses of Lemma B.7. We have already satisfied the conditions on $\varepsilon$ and $q$. We notice that (B.2) implies that it suffices to enforce simply $N \geq L^{10}$, and following Lemma C.4, we can bound $C_\mathcal{M}$ as in (B.62) in the proof of Lemma B.7 by

$$C_\mathcal{M} \leq 1 + \frac{\text{len}(\mathcal{M}_+)}{\mu^\infty(\mathcal{M}_+)} + \frac{\text{len}(\mathcal{M}_-)}{\mu^\infty(\mathcal{M}_-)} \leq 2\frac{1 + \rho_{\max}}{\rho_{\min}}.$$

Because $n \geq L^{99}$ and $L \geq C_\rho(1 + \rho_{\max})$, we can eliminate $C_\mathcal{M}$ from the lower bound on $d$ while paying only an extra factor of 2 in the constant. In addition, because $\kappa \geq 1$ and $C_\lambda \geq \max\{1, 1/c_\lambda\}$, we can remove the $\kappa^{2/5}$ and $\left(\frac{\kappa}{c_\lambda}\right)^{1/3}$ lower bounds on $n$, since they are enforced through $L$ already via the bound $L \geq K''\kappa^2 C_\lambda$, worsening the absolute constant if needed. These simplifications lead us to the sufficient conditions (plus the certificate existence hypotheses)

$$d \geq K \max\{\log(1/\delta), \log(nn_0)\}$$
$$n \geq K'L^{99}d^9 \log^9 L$$
$$L \geq K'' \max\left\{\frac{C_\rho^{11+22q_{\text{cert}}/3}(1 + \rho_{\max})^{11/2}d^{33/4}\log^{44/3} L}{(\min\{\mu^\infty(\mathcal{M}_+), \mu^\infty(\mathcal{M}_-)\})^{11/2}}, \kappa^2 C_\lambda\right\}$$
$$N \geq L^{10}.$$

We ignore the condition on $N$ below, since it matches with the theorem statement. When $\delta \leq 1/e$, given that $n_0 \geq 3$ we have $nn_0 \geq e$ and $\max\{\log(1/\delta), \log(nn_0)\} \leq \log(1/\delta)\log(nn_0)$. For the sake of simplicity, we can also round up the fractional constants in the lower bound on $L$. We can eliminate $d$ from these sufficient conditions by substituting the lower bound into the conditions on $n$ and $L$, and this also implies that our conditions on certificate existence in the theorem statement suffice for the certificate existence hypothesis for Lemma B.7. Thus, we have the remaining sufficient conditions

$$n \geq KL^{99}\log^9(1/\delta)\log^9(nn_0)\log^9 L$$
$$L \geq K' \max\left\{\frac{C_\rho^{11+8q_{\text{cert}}}(1 + \rho_{\max})^6\log^9(1/\delta)\log^9(nn_0)\log^{15} L}{(\min\{\mu^\infty(\mathcal{M}_+), \mu^\infty(\mathcal{M}_-)\})^{11/2}}, \kappa^2 C_\lambda\right\}.$$

Using Lemma B.15 and choosing $L$ larger than a sufficiently large absolute constant and larger than $\log(1/\delta)$, we obtain that it suffices to enforce for $n$

$$n \geq KL^{99}\log^9(1/\delta)\log^{18}(Ln_0).$$

In the hypotheses of the theorem, we have chosen the equality $n = KL^{99}\log^9(1/\delta)\log^{18}(Ln_0)$ in the last bound. This implies $\log(nn_0) \leq C\log(Ln_0)$, so it suffices to enforce the $L$ lower bound

$$L \geq K' \max\left\{\frac{C_\rho^{11+8q_{\text{cert}}}(1 + \rho_{\max})^6\log^9(1/\delta)\log^{24}(Ln_0)}{(\min\{\mu^\infty(\mathcal{M}_+), \mu^\infty(\mathcal{M}_-)\})^{11/2}}, \kappa^2 C_\lambda\right\}.$$

Defining, as in the theorem

$$C_{\mu^\infty} = \frac{C_\rho^{11+8q_{\text{cert}}}(1 + \rho_{\max})^6}{(\min\{\mu^\infty(\mathcal{M}_+), \mu^\infty(\mathcal{M}_-)\})^{11/2}},$$

and using $C_{\mu^\infty} \geq 1$, we can worsen the absolute constant $K'$ in order to apply Lemma B.15 once again, obtaining the simplified condition

$$L \geq CK' \max\{C_{\mu^\infty}\log^9(1/\delta)\log^{24}(C_{\mu^\infty}n_0\log(1/\delta)), \kappa^2 C_\lambda\}.$$

These conditions reflect what is stated in the lemma. $\qquad\square$

**Theorem B.2.** *Let $\mathcal{M}$ be a $d_0$-dimensional Riemannian submanifold of $\mathbb{S}^{n_0-1}$. For any $d \geq Kd_0\log(nn_0 C_\mathcal{M})$, if $n \geq K'd^4 L$ then one has on an event of probability at least $1 - e^{-cd}$*

$$\sup_{(\boldsymbol{x}, \boldsymbol{x}')\in\mathcal{M}\times\mathcal{M}} \left|\Theta(\boldsymbol{x}, \boldsymbol{x}') - \frac{n}{2}\sum_{\ell=0}^{L-1}\cos\left(\varphi^{(\ell)}(\nu)\right)\prod_{\ell'=\ell}^{L-1}\left(1 - \frac{\varphi^{(\ell')}(\nu)}{\pi}\right)\right| \leq \sqrt{d^4 nL^3},$$

*where we write $\nu = \angle(\boldsymbol{x}, \boldsymbol{x}')$ in context with an abuse of notation, $c, K, K' > 0$ are absolute constants, and $C_\mathcal{M} > 0$ depends only on the number of connected components of $\mathcal{M}$ and their diameters and curvatures (Lemma C.4).*

*Proof.* We have by the definition of $\Theta$

$$\Theta(\boldsymbol{x}, \boldsymbol{x}') = \langle \boldsymbol{\alpha}^L(\boldsymbol{x}), \boldsymbol{\alpha}^L(\boldsymbol{x}') \rangle + \sum_{\ell=0}^{L-1} \langle \boldsymbol{\alpha}^\ell(\boldsymbol{x}), \boldsymbol{\alpha}^\ell(\boldsymbol{x}') \rangle \langle \boldsymbol{\beta}^\ell(\boldsymbol{x}), \boldsymbol{\beta}^\ell(\boldsymbol{x}') \rangle. \tag{B.4}$$

Under the stated hypotheses, Lemmas D.10 and D.13 give uniform control of each of the terms appearing in this expression with suitable probability to tolerate $2L + 1$ union bounds, which gives simultaneous uniform control of the factors on an event $\mathcal{E}$ with probability at least $1 - e^{-cd}$. Starting from (B.4), we can write with the triangle inequality

$$\left| \Theta(\boldsymbol{x}, \boldsymbol{x}') - \frac{n}{2} \sum_{\ell=0}^{L-1} \cos\left(\varphi^{(\ell)}(\nu)\right) \prod_{\ell'=\ell}^{L-1} \left(1 - \frac{\varphi^{(\ell')}(\nu)}{\pi}\right) \right| \leq \left| \langle \boldsymbol{\alpha}^L(\boldsymbol{x}), \boldsymbol{\alpha}^L(\boldsymbol{x}') \rangle \right|$$
$$+ \sum_{\ell=0}^{L-1} \left| \langle \boldsymbol{\alpha}^\ell(\boldsymbol{x}), \boldsymbol{\alpha}^\ell(\boldsymbol{x}') \rangle \langle \boldsymbol{\beta}^\ell(\boldsymbol{x}), \boldsymbol{\beta}^\ell(\boldsymbol{x}') \rangle - \frac{n}{2} \sum_{\ell=0}^{L-1} \cos\left(\varphi^{(\ell)}(\nu)\right) \prod_{\ell'=\ell}^{L-1} \left(1 - \frac{\varphi^{(\ell')}(\nu)}{\pi}\right) \right|. \tag{B.5}$$

By the triangle inequality, we have

$$\left| \langle \boldsymbol{\alpha}^\ell(\boldsymbol{x}), \boldsymbol{\alpha}^\ell(\boldsymbol{x}') \rangle \langle \boldsymbol{\beta}^\ell(\boldsymbol{x}), \boldsymbol{\beta}^\ell(\boldsymbol{x}') \rangle - \frac{n}{2} \cos\left(\varphi^{(\ell)}(\nu)\right) \prod_{\ell'=\ell}^{L-1} \left(1 - \frac{\varphi^{(\ell')}(\nu)}{\pi}\right) \right|$$
$$\leq \left| \langle \boldsymbol{\alpha}^\ell(\boldsymbol{x}), \boldsymbol{\alpha}^\ell(\boldsymbol{x}') \rangle \right| \left| \langle \boldsymbol{\beta}^\ell(\boldsymbol{x}), \boldsymbol{\beta}^\ell(\boldsymbol{x}') \rangle - \frac{n}{2} \prod_{\ell'=\ell}^{L-1} \left(1 - \frac{\varphi^{(\ell')}(\nu)}{\pi}\right) \right|$$
$$+ \left| \frac{n}{2} \prod_{\ell'=\ell}^{L-1} \left(1 - \frac{\varphi^{(\ell')}(\nu)}{\pi}\right) \right| \left| \langle \boldsymbol{\alpha}^\ell(\boldsymbol{x}), \boldsymbol{\alpha}^\ell(\boldsymbol{x}') \rangle - \cos\left(\varphi^{(\ell)}(\nu)\right) \right|.$$

Under the conditions on $n$, $L$, and $d$, we have on the event $\mathcal{E}$ that for each $\ell$

$$\sup_{(\boldsymbol{x}, \boldsymbol{x}') \in \mathcal{M}} \left| \langle \boldsymbol{\alpha}^\ell(\boldsymbol{x}), \boldsymbol{\alpha}^\ell(\boldsymbol{x}') \rangle \right| \leq 2,$$

so we can conclude that on $\mathcal{E}$

$$\left| \langle \boldsymbol{\alpha}^\ell(\boldsymbol{x}), \boldsymbol{\alpha}^\ell(\boldsymbol{x}') \rangle \langle \boldsymbol{\beta}^\ell(\boldsymbol{x}), \boldsymbol{\beta}^\ell(\boldsymbol{x}') \rangle - \frac{n}{2} \cos\left(\varphi^{(\ell)}(\nu)\right) \prod_{\ell'=\ell}^{L-1} \left(1 - \frac{\varphi^{(\ell')}(\nu)}{\pi}\right) \right| \leq 3\sqrt{d^4 nL}.$$

The conditions on $n$, $d$, and $L$ imply that this residual is larger than that incurred by the level-$L$ features, which is no larger than 2. Returning to (B.5), we have shown that on $\mathcal{E}$

$$\left| \Theta(\boldsymbol{x}, \boldsymbol{x}') - \frac{n}{2} \sum_{\ell=0}^{L-1} \cos\left(\varphi^{(\ell)}(\nu)\right) \prod_{\ell'=\ell}^{L-1} \left(1 - \frac{\varphi^{(\ell')}(\nu)}{\pi}\right) \right| \leq C\sqrt{d^4 nL^3}.$$

After adjusting the other absolute constants to absorb $C$ into $d$, this gives the claim. $\qquad\square$

**Theorem B.3** (Pointwise Version of Theorem B.2). *Let $\mathcal{M}$ be a $d_0$-dimensional Riemannian submanifold of $\mathbb{S}^{n_0-1}$. For any $d \geq K \log n$, if $n \geq K' \max\{1, d^4 L\}$ then one has for any $(\boldsymbol{x}, \boldsymbol{x}') \in \mathcal{M} \times \mathcal{M}$*

$$\mathbb{P}\left[ \left| \Theta(\boldsymbol{x}, \boldsymbol{x}') - \frac{n}{2} \sum_{\ell=0}^{L-1} \cos\left(\varphi^{(\ell)}(\nu)\right) \prod_{\ell'=\ell}^{L-1} \left(1 - \frac{\varphi^{(\ell')}(\nu)}{\pi}\right) \right| \leq \sqrt{d^4 nL^3} \right] \geq 1 - e^{-cd},$$

*where we write $\nu = \angle(\boldsymbol{x}, \boldsymbol{x}')$ in context with an abuse of notation, and $c, K, K' > 0$ are absolute constants.*

*Proof.* Follow the proof of Theorem B.2, but invoke the pointwise versions of the uniform concentration results used there (i.e., Lemmas D.1 and D.4) after rescaling $d$ to relocate the $\log n$ terms. $\quad\square$

**Proposition B.4.** *Let $\mathcal{M}$ be an $r$-instance of the two circles geometry studied in Appendix C.1.1, with $r \geq 1/2$. For any $0 < \delta \leq 1/e$, if $n \geq KL^5 \log^4(1/\delta) \log^4(Ln_0 \log(1/\delta))$ and $L \geq K'(1 - r^2)^{-1/2}$, then there exist absolute constants $C_5, C_6 > 0$ and a function $g$ such that (B.1) is satisfied with the choice $q_{\mathrm{cert}} = 1/2$ with probability at least $1 - 3\delta$. The constants $K, K' > 0$ are absolute.*

*Proof.* Given $r \geq \frac{1}{2}$ and $L \geq \max\{K, (\pi/2)(1-r^2)^{-1/2}\}$, we have by Lemma C.1 that there exists $g$ such that $\int_{\mathcal{M}} \psi \circ \angle(\,\cdot\,, \boldsymbol{x}') g(\boldsymbol{x}') \, d\mu^{\infty}(\boldsymbol{x}') = \hat{\zeta}$, with

$$\|g\|_{L^2_{\mu}\infty} \leq (64/\sqrt{\pi}) \frac{\|\hat{\zeta}\|_{L^{\infty}(\mathcal{M})}}{n\rho_{\min}^{1/2}}. \tag{B.6}$$

By this bound, the triangle inequality, the Minkowski inequality, and the fact that $\mu^{\infty}$ is a probability measure, we have

$$\|\boldsymbol{\Theta}[g] - \zeta\|_{L^2_{\mu}\infty} \leq \|\boldsymbol{\Theta} - \psi \circ \angle\|_{L^{\infty}(\mathcal{M} \times \mathcal{M})} \|g\|_{L^2_{\mu}\infty} + \|\zeta - \hat{\zeta}\|_{L^2_{\mu}\infty}$$

$$\leq C\|\boldsymbol{\Theta} - \psi \circ \angle\|_{L^{\infty}(\mathcal{M} \times \mathcal{M})} \frac{\|\hat{\zeta}\|_{L^{\infty}(\mathcal{M})}}{n\rho_{\min}^{1/2}} + \|\zeta - \hat{\zeta}\|_{L^{\infty}(\mathcal{M})}. \tag{B.7}$$

An application of Theorem B.2 and Lemma C.11 gives that on an event of probability at least $1 - e^{-cd}$

$$\|\boldsymbol{\Theta} - \psi \circ \angle\|_{L^{\infty}(\mathcal{M} \times \mathcal{M})} \leq Cn/L$$

if $d \geq Kd_0 \log(nn_0 C_{\mathcal{M}})$ and $n \geq K'd^4 L^5$. An application of Lemma D.11 gives

$$\mathbb{P}\left[\left\|\hat{\zeta} - \zeta\right\|_{L^{\infty}(\mathcal{M})} \leq \frac{\sqrt{2d}}{L}\right] \geq 1 - e^{-cd}$$

and

$$\mathbb{P}\left[\|\zeta\|_{L^{\infty}(\mathcal{M})} \leq \sqrt{d}\right] \geq 1 - e^{-cd}$$

as long as $n \geq Kd^4 L^5$ and $d \geq K'd_0 \log(nn_0 C_{\mathcal{M}})$, where we use these conditions to simplify the residual that appears in Lemma D.11. In particular, combining the previous two bounds with the triangle inequality and a union bound and then rescaling $d$, which worsens the constant $c$ and the absolute constants in the preceding conditions, gives

$$\mathbb{P}\left[\left\|\hat{\zeta}\right\|_{L^{\infty}(\mathcal{M})} \leq \sqrt{d}\right] \geq 1 - 2e^{-cd}.$$

Combining these bounds using a union bound and substituting into (B.7), we get that under the preceding conditions, on an event of probability at least $1 - 3e^{-cd}$ we have

$$\|\boldsymbol{\Theta}[g] - \zeta\|_{L^2_{\mu}\infty} \leq \frac{C\sqrt{d}}{L}\left(1 + \frac{1}{\rho_{\min}^{1/2}}\right)$$

$$\leq \frac{C\sqrt{d}}{L} \max\{\rho_{\min}^{1/2}, \rho_{\min}^{-1/2}\},$$

where we worst-case the density constant in the second line, and in addition, on the same event, we have by (B.6)

$$\|g\|_{L^2_{\mu}\infty} \leq (64/\sqrt{\pi}) \frac{\sqrt{d}}{n\rho_{\min}^{1/2}}.$$

To conclude, we simplify the preceding conditions on $n$ and turn the parameter $d$ into a parameter $\delta > 0$ in order to obtain the claimed form of the result. We have in this setting $d_0 = 1$, and also that $C_{\mathcal{M}}$ is bounded by an absolute constant; since $n_0 \geq 3$, we can thus eliminate the parameter $C_{\mathcal{M}}$ from our hypotheses by adding an extra absolute constant factor. Choosing $d \geq (1/c) \log(1/\delta)$, we obtain that the previous two bounds hold on an event of probability at least $1 - 3\delta$. When $\delta \leq 1/e$, given that $n_0 \geq 3$ we have $nn_0 \geq e$ and $\max\{\log(1/\delta), \log(nn_0)\} \leq \log(1/\delta) \log(nn_0)$, so that it suffices to enforce the requirement $d \geq K \log(1/\delta) \log(nn_0)$ for a certain absolute constant $K > 0$. We can then substitute this lower bound on $d$ into the two certificate bounds above to obtain the form claimed in (B.1) with $q_{\text{cert}} = 1/2$. For the hypothesis on $n$, we substitute this lower bound on $d$ into the condition on $n$ to obtain the sufficient condition $n \geq K'L^5 \log^4(1/\delta) \log^4(nn_0)$. Using Lemma B.15 and possibly worsening absolute constants, we then get that it suffices to enforce $n \geq K'L^5 \log^4(1/\delta) \log^4(Ln_0 \log(1/\delta))$, which is the hypothesis in the result. $\qquad\square$

**Theorem B.5.** *There exist absolute constants $c, C, K, K' > 0$ such that for any $d \geq Kn_0 \log n$, if $n \geq K'd^4 L$, then on an event of probability at least $1 - e^{-cd}$ the natural extension of $f_{\boldsymbol{\theta}_0}$ to $\mathbb{R}^{n_0}$ is $3\sqrt{d}$-Lipschitz.*

*Proof.* The proof is a simple application of Lemma B.17, which (because $f_{\boldsymbol{\theta}_0}$ is 1-nonnegatively homogeneous and so are all its intermediate feature maps $\boldsymbol{\alpha}_{\boldsymbol{\theta}_0}^\ell(\boldsymbol{x})$) implies that it suffices to control the Lipschitz constants of the maps and bound them on the unit sphere, together with Lemmas D.11 and D.12. In particular, for any $d \geq Kn_0 \log(n)$ and any $n \geq K'd^4 L$, we have that there exists an event of probability at least $1 - e^{-cd}$ on which

$$\|f_{\boldsymbol{\theta}}\|_{L^\infty(\mathbb{S}^{n_0-1})} \leq \sqrt{d},$$

and

$$\left\|f_{\boldsymbol{\theta}}\big|_{\mathbb{S}^{n_0-1}}\right\|_{\mathrm{Lip}} \leq \sqrt{d}.$$

Applying Lemma B.17, it follows that $f_{\boldsymbol{\theta}_0} : \mathbb{R}^{n_0} \to \mathbb{R}$ is $3\sqrt{d}$-Lipschitz on an event of probability at least $1 - e^{-cd}$.

$\square$

## B.2 SUPPORTING RESULTS ON DYNAMICS

**Lemma B.6** (Nominal). *Suppose $C_{\mathrm{err}}, C_{\mathrm{cert}}, q_{\mathrm{cert}} > 0$ are absolute constants. Then there exist absolute constants $c, c', C', C'', C''' > 0$ and absolute constants $K, K', K'' > 0$ such that for any $d \geq Kd_0 \log(nn_0 C_{\mathcal{M}})$ and any $1/2 \leq q \leq 1$, if $n \geq K'd^4 L^5$, if $L \geq K''dC_\rho^{2q_{\mathrm{cert}}}$, and if additionally there exists $g \in L^2_{\mu\infty}(\mathcal{M})$ satisfying*

$$\|\boldsymbol{\Theta}[g] - \zeta\|_{L^2_{\mu\infty}(\mathcal{M})} \leq C_{\mathrm{err}} C_\rho^{q_{\mathrm{cert}}} \frac{\sqrt{d}}{L}; \qquad \|g\|_{L^2_{\mu\infty}(\mathcal{M})} \leq C_{\mathrm{cert}} \rho_{\min}^{-q_{\mathrm{cert}}} \frac{\sqrt{d}}{n}$$

*and $\tau > 0$ is chosen such that*

$$\tau \leq \frac{c'}{nL},$$

*then one has*

$$\mathbb{P}\left[\bigcap_{0 \leq k \leq L^q/(n\tau)} \left\{\|\zeta_k^\infty\|_{L^2_{\mu\infty}(\mathcal{M})} \leq \sqrt{d}\right\}\right] \geq 1 - e^{-cd},$$

*and in addition*

$$\mathbb{P}\left[\bigcap_{C'\sqrt{d}/(n\tau\rho_{\min}^{q_{\mathrm{cert}}}) \leq k \leq L^q/(n\tau)} \left\{\|\zeta_k^\infty\|_{L^2_{\mu\infty}(\mathcal{M})} \leq \frac{C'' C_\rho^{q_{\mathrm{cert}}} \sqrt{d} \log L}{nk\tau}\right\}\right] \geq 1 - e^{-cd}.$$

*Moreover, one has*

$$\mathbb{P}\left[\bigcap_{0 \leq k \leq L^q/(n\tau)} \left\{\sum_{s=0}^k \|\zeta_s^\infty\|_{L^2_{\mu\infty}(\mathcal{M})} \leq C_\rho^{2q_{\mathrm{cert}}} \frac{C''' d \log^2 L}{n\tau}\right\}\right] \geq 1 - e^{-cd}.$$

*The constant $C_\rho = \max\{\rho_{\min}, \rho_{\min}^{-1}\}$.*

*Proof.* We will combine Lemma B.12 with various probabilistic results to obtain a simple final form for the bound from this result.

Invoking Lemma B.12, we can assert that for any step size $\tau > 0$ satisfying

$$\tau < \frac{1}{\|\boldsymbol{\Theta}\|_{L^2_{\mu\infty}(\mathcal{M}) \to L^2_{\mu\infty}(\mathcal{M})}}, \tag{B.8}$$

and for any $k$ satisfying

$$k\tau \geq \sqrt{\frac{3e}{2}} \frac{\|g\|_{L^2_{\mu\infty}(\mathcal{M})}}{\|\zeta\|_{L^\infty(\mathcal{M})}}, \tag{B.9}$$

the population dynamics satisfy

$$\|\zeta_k^\infty\|_{L_\mu^2\infty(\mathcal{M})} \le \sqrt{3}\|\Theta[g] - \zeta\|_{L_\mu^2\infty(\mathcal{M})} - \frac{3\|g\|_{L_\mu^2\infty(\mathcal{M})}}{k\tau} \log\left(\sqrt{\frac{3}{2}}\frac{\|g\|_{L_\mu^2\infty(\mathcal{M})}}{\|\zeta\|_{L^\infty(\mathcal{M})}k\tau}\right). \quad \text{(B.10)}$$

We state the bounds we will apply to simplify this expression. An application of Lemma D.11 gives

$$\mathbb{P}\left[\left\|\hat\zeta - \zeta\right\|_{L^\infty(\mathcal{M})} \le \frac{\sqrt{2d}}{L}\right] \ge 1 - e^{-cd} \quad \text{(B.11)}$$

and

$$\mathbb{P}\left[\|\zeta\|_{L^\infty(\mathcal{M})} \le \sqrt{d}\right] \ge 1 - e^{-cd} \quad \text{(B.12)}$$

as long as $n \ge Kd^4L^5$ and $d \ge K'd_0\log(nn_0C_\mathcal{M})$, where we use these conditions to simplify the residual that appears in the version of (B.11) quoted in Lemma D.11. In particular, combining (B.11) and (B.12) with the triangle inequality and a union bound and then rescaling $d$, which worsens the constant $c$ and the absolute constants in the preceding conditions, gives

$$\mathbb{P}\left[\left\|\hat\zeta\right\|_{L^\infty(\mathcal{M})} \le \sqrt{d}\right] \ge 1 - 2e^{-cd}. \quad \text{(B.13)}$$

In addition, we can write using the triangle inequality

$$\|\zeta\|_{L^\infty(\mathcal{M})} \ge \left\|\hat\zeta\right\|_{L^\infty(\mathcal{M})} - \left\|\zeta - \hat\zeta\right\|_{L^\infty(\mathcal{M})},$$

and

$$\begin{aligned}
\left\|\hat\zeta\right\|_{L^\infty(\mathcal{M})} &= \sup_{\boldsymbol{x}\in\mathcal{M}}\left|f_\star(\boldsymbol{x}) - \int_\mathcal{M} f_{\boldsymbol{\theta}_0}(\boldsymbol{x}')\,\mathrm{d}\mu^\infty(\boldsymbol{x}')\right| \\
&= \max\left\{\left|\int_\mathcal{M} f_{\boldsymbol{\theta}_0}(\boldsymbol{x}')\,\mathrm{d}\mu^\infty(\boldsymbol{x}') - 1\right|, \left|\int_\mathcal{M} f_{\boldsymbol{\theta}_0}(\boldsymbol{x}')\,\mathrm{d}\mu^\infty(\boldsymbol{x}') + 1\right|\right\} \\
&\ge 1,
\end{aligned}$$

so that, by (B.11), we have if $L \ge 2\sqrt{d}$

$$\mathbb{P}\left[\|\zeta\|_{L^\infty(\mathcal{M})} \ge \frac{1}{2}\right] \ge 1 - e^{-cd}. \quad \text{(B.14)}$$

Because $\mu^\infty$ is a probability measure, Jensen's inequality, the Schwarz inequality, and the triangle inequality give

$$\begin{aligned}
\|\boldsymbol{\Theta}\|_{L_\mu^2\infty(\mathcal{M})\to L_\mu^2\infty(\mathcal{M})} &\le \sup_{(\boldsymbol{x},\boldsymbol{x}')\in\mathcal{M}\times\mathcal{M}} |\Theta(\boldsymbol{x},\boldsymbol{x}')| \\
&\le \sup_{(\boldsymbol{x},\boldsymbol{x}')\in\mathcal{M}\times\mathcal{M}} |\Theta(\boldsymbol{x},\boldsymbol{x}') - \psi_1\circ\angle(\boldsymbol{x},\boldsymbol{x}')| \\
&\quad + \sup_{(\boldsymbol{x},\boldsymbol{x}')\in\mathcal{M}\times\mathcal{M}} |\psi_1\circ\angle(\boldsymbol{x},\boldsymbol{x}')|,
\end{aligned}$$

and an application of Theorem B.2 and Lemma E.5 then gives that on an event of probability at least $1 - e^{-cd}$

$$\|\boldsymbol{\Theta}\|_{L_\mu^2\infty(\mathcal{M})\to L_\mu^2\infty(\mathcal{M})} \le CnL \quad \text{(B.15)}$$

provided $d \ge Kd_0\log(nn_0C_\mathcal{M})$ and $n \ge K'd^4L$. We will write $\mathcal{E}$ for the event consisting of the union of the events invoked for the bounds (B.11) to (B.15), which has probability at least $1 - e^{-cd}$ by a union bound and a choice of $d \ge K$. We will conclude by simplifying (B.10) on $\mathcal{E}$. First, we note that by (B.15), the step size condition (B.8) is satisfied on $\mathcal{E}$ provided

$$\tau \le \frac{c}{nL}, \quad \text{(B.16)}$$

which holds under our hypotheses. Next, on $\mathcal{E}$, we write using decreasingness of $x \mapsto -\log x$ and (B.12)

$$\begin{aligned}
-\frac{3\|g\|_{L_\mu^2\infty(\mathcal{M})}}{k\tau}\log\left(\sqrt{\frac{3}{2}}\frac{\|g\|_{L_\mu^2\infty(\mathcal{M})}}{\|\zeta\|_{L^\infty(\mathcal{M})}k\tau}\right) &\le -\frac{3\|g\|_{L_\mu^2\infty(\mathcal{M})}}{k\tau}\log\left(\sqrt{\frac{3}{2}}\frac{\|g\|_{L_\mu^2\infty(\mathcal{M})}}{k\tau\sqrt{d}}\right) \\
&= -\sqrt{6d}\frac{\sqrt{3}\|g\|_{L_\mu^2\infty(\mathcal{M})}}{\sqrt{2}k\tau\sqrt{d}}\log\left(\sqrt{\frac{3}{2}}\frac{\|g\|_{L_\mu^2\infty(\mathcal{M})}}{k\tau\sqrt{d}}\right).
\end{aligned}$$

$$\text{(B.17)}$$

By the hypothesis on $g$, we have on $\mathcal{E}$

$$\|g\|_{L^2_{\mu\infty}(\mathcal{M})} \leq C\rho_{\min}^{-q_{\mathrm{cert}}} \frac{\sqrt{d}}{n}, \tag{B.18}$$

and so it follows that on $\mathcal{E}$

$$\sqrt{\frac{3}{2}} \frac{\|g\|_{L^2_{\mu\infty}(\mathcal{M})}}{k\tau\sqrt{d}} \leq \frac{C}{nk\tau\rho_{\min}^{q_{\mathrm{cert}}}}.$$

The function $x \mapsto -x\log x$ is a strictly increasing function on $[0, e^{-1}]$, so when $k$ is chosen such that

$$\frac{Ce}{n\tau\rho_{\min}^{q_{\mathrm{cert}}}} \leq k, \tag{B.19}$$

we have on $\mathcal{E}$ by (B.17)

$$-\frac{3\|g\|_{L^2_{\mu\infty}(\mathcal{M})}}{k\tau} \log\left(\sqrt{\frac{3}{2}} \frac{\|g\|_{L^2_{\mu\infty}(\mathcal{M})}}{\|\zeta\|_{L^\infty(\mathcal{M})}k\tau}\right) \leq \frac{C\sqrt{6d}}{nk\tau\rho_{\min}^{q_{\mathrm{cert}}}} \log\left(C^{-1}nk\tau\rho_{\min}^{q_{\mathrm{cert}}}\right). \tag{B.20}$$

Additionally, in the context of the condition (B.9), notice that by (B.14) and (B.18), on $\mathcal{E}$ we have

$$\sqrt{\frac{3e}{2}} \frac{\|g\|_{L^2_{\mu\infty}(\mathcal{M})}}{\tau\|\zeta\|_{L^\infty(\mathcal{M})}} \leq \frac{Ce\sqrt{d}}{n\tau\rho_{\min}^{q_{\mathrm{cert}}}},$$

so that given $d \geq 1$, we have that the choice

$$k \geq \frac{Ce\sqrt{d}}{n\tau\rho_{\min}^{q_{\mathrm{cert}}}} \tag{B.21}$$

implies both conditions (B.9) and (B.19). We can simplify (B.20) using the hypothesis $k\tau \leq L^q/n$ with $1/2 \leq q \leq 1$: we get

$$\frac{nk\tau\rho_{\min}^{q_{\mathrm{cert}}}}{C} \leq \frac{L^q\rho_{\min}^{q_{\mathrm{cert}}}}{C} \leq L^{1+q},$$

where the last inequality requires $L \geq \rho_{\min}^{q_{\mathrm{cert}}}/C$, which implies

$$-\frac{3\|g\|_{L^2_{\mu\infty}(\mathcal{M})}}{k\tau} \log\left(\sqrt{\frac{3}{2}} \frac{\|g\|_{L^2_{\mu\infty}(\mathcal{M})}}{\|\zeta\|_{L^\infty(\mathcal{M})}k\tau}\right) \leq \frac{C'\sqrt{d}\log L}{nk\tau\rho_{\min}^{q_{\mathrm{cert}}}}. \tag{B.22}$$

The conditions we need to satisfy on $k\tau$ can be stated together as

$$\frac{Ce\sqrt{d}}{n\rho_{\min}^{q_{\mathrm{cert}}}} \leq k\tau \leq L^q/n,$$

and it is possible to satisfy these conditions simultaneously as long as

$$L \geq \left(\frac{Ce\sqrt{d}}{\rho_{\min}^{q_{\mathrm{cert}}}}\right)^{1/q}.$$

We obtain an upper bound $\frac{C^2e^2d}{\rho_{\min}^{2q_{\mathrm{cert}}}}$ for the quantity on the RHS of this inequality from $q \geq 1/2$; it suffices to choose $L$ larger than this upper bound instead. The other simplifications are easier: using the assumption on the norm of $\Theta[g] - \zeta$, we have

$$\|\Theta[g] - \zeta\|_{L^2_{\mu\infty}(\mathcal{M})} \leq C_\rho^{q_{\mathrm{cert}}} \frac{C\sqrt{d}}{L\rho_{\min}^{1/2}}.$$

Worst-casing terms using our hypotheses on $d$ and $L$ to obtain a simplified bound, on $\mathcal{E}$, we have thus shown that when (B.21) is satisfied, we have

$$\|\zeta_k^\infty\|_{L^2_{\mu\infty}(\mathcal{M})} \leq CC_\rho^{q_{\mathrm{cert}}}\sqrt{d}\left(\frac{1}{L} + \frac{\log L}{nk\tau}\right).$$

We have

$$\frac{1}{L} \le \frac{\log L}{nk\tau} \iff \frac{L\log L}{n} \ge k\tau,$$

which is implied by the hypothesis $k\tau \le L^q/n$ as long as $L \ge e$. So we can simplify to

$$\|\zeta_k^\infty\|_{L^2_{\mu\infty}(\mathcal{M})} \le \frac{CC_\rho^{q_{\mathrm{cert}}}\sqrt{d}\log L}{nk\tau}.$$

We also need a bound that works for $k$ that do not satisfy (B.21). From the update equation for the dynamics in the proof of Lemma B.12 and the choice of $\tau$, we also have

$$\|\zeta_k^\infty\|_{L^2_{\mu\infty}(\mathcal{M})} \le \|\zeta\|_{L^2_{\mu\infty}(\mathcal{M})} \le \sqrt{d},$$

where the last bound is valid on $\mathcal{E}$. Finally, we can obtain the claimed sum bound by calculating using our 'small-$k$' and 'large-$k$' bounds:

$$\sum_{s=0}^{k}\|\zeta_s^\infty\|_{L^2_{\mu\infty}(\mathcal{M})} = \sum_{s=0}^{\lfloor C\sqrt{d}/(n\tau\rho_{\min}^{q_{\mathrm{cert}}})\rfloor}\|\zeta_s^\infty\|_{L^2_{\mu\infty}(\mathcal{M})} + \sum_{s=\lceil C\sqrt{d}/(n\tau\rho_{\min}^{q_{\mathrm{cert}}})\rceil}^{k}\|\zeta_s^\infty\|_{L^2_{\mu\infty}(\mathcal{M})}$$

$$\le \sqrt{d}\left(1 + \frac{C'\sqrt{d}}{n\tau\rho_{\min}^{q_{\mathrm{cert}}}}\right) + \frac{C''C_\rho^{q_{\mathrm{cert}}}\sqrt{d}\log L}{n\tau}\sum_{s=\lceil C\sqrt{d}/(n\tau\rho_{\min}^{q_{\mathrm{cert}}})\rceil}^{k}\frac{1}{s}$$

$$\le \frac{C'd}{n\tau\rho_{\min}^{q_{\mathrm{cert}}}} + \frac{C''C_\rho^{q_{\mathrm{cert}}}\sqrt{d}\log L}{n\tau}\left(\frac{n\tau\rho_{\min}^{q_{\mathrm{cert}}}}{C\sqrt{d}} + \int_{C\sqrt{d}/(n\tau\rho_{\min}^{q_{\mathrm{cert}}})}^{k}\frac{ds}{s}\right)$$

$$\le \frac{Cd}{n\tau\rho_{\min}^{q_{\mathrm{cert}}}} + C'\max\left\{\rho_{\min}^{2q_{\mathrm{cert}}}, 1\right\}\log L + \frac{C''C_\rho^{q_{\mathrm{cert}}}\sqrt{d}\log^2 L}{n\tau},$$

where the second inequality uses standard estimates for the harmonic numbers and $C'\sqrt{d}/(n\tau\rho_{\min}^{q_{\mathrm{cert}}}) \ge 1$, which follows from $\tau \le c'/(nL)$, $d \ge 1$ and $L \ge K\rho_{\min}^{q_{\mathrm{cert}}}$ for a suitable absolute constant $K$; and the third inequality integrates and simplifies, using $k\tau \le L/n$ and again $d \ge 1$ and $L \ge C\rho_{\min}^{q_{\mathrm{cert}}}$. Worst-casing constants and using $n\tau \le 1$, we simplify this last bound to

$$\sum_{s=0}^{k}\|\zeta_s^\infty\|_{L^2_{\mu\infty}(\mathcal{M})} \le \max\left\{\rho_{\min}^{2q_{\mathrm{cert}}}, \frac{1}{\rho_{\min}^{2q_{\mathrm{cert}}}}\right\}\frac{Cd\log^2 L}{n\tau}.$$

To see that the conditions on $L$ in the statement of the result suffice, note that we have to satisfy (say) $L \ge K\rho_{\min}^{q_{\mathrm{cert}}}$ and $L \ge K'\rho_{\min}^{-q_{\mathrm{cert}}}$; the first of these lower bounds is tighter when $\rho_{\min} \ge 1$, and the second when $\rho_{\min} < 1$, and so it suffices to require $L \ge K\rho_{\min}^{2q_{\mathrm{cert}}}$ and $L \ge K'\rho_{\min}^{-2q_{\mathrm{cert}}}$ instead. $\qquad\square$

**Lemma B.7** (Nominal to Finite). *Let $d_0 = 1$, and suppose $C_{\mathrm{err}}, C_{\mathrm{cert}}, q_{\mathrm{cert}} > 0$ are absolute constants. Then there exist absolute constants $c, c', C', C'', C''' > 0$ and absolute constants $K, K', K'', K''' > 0$ such that for any $d \ge Kd_0\log(nn_0C_\mathcal{M})$, any $1/2 \le q < 1$ and any $0 < \delta \le 1$, if $L \ge K'\max\{C_\rho^{2q_{\mathrm{cert}}}d, \kappa^2 C_\lambda\}$, if*

$$n \ge K''\max\left\{e^{252/\delta}L^{60+44q}d^9\log^9 L, \kappa^{2/5}, \left(\frac{\kappa}{c_\lambda}\right)^{1/3}\right\},$$

*and if*

$$N^{1/(2+\delta)} \ge K'''\frac{C_\rho^{7/2+8q_{\mathrm{cert}}/3}(1+\rho_{\max})^{7/6}e^{119/(3\delta)}}{\min\left\{\mu^\infty(\mathcal{M}_+)^{1/2}, \mu^\infty(\mathcal{M}_-)^{1/2}\right\}}d^{5/4}L^{5/2+2q}\log L,$$

*and if additionally there exists $g \in L^2_{\mu\infty}(\mathcal{M})$ satisfying*

$$\|\Theta[g] - \zeta\|_{L^2_{\mu\infty}(\mathcal{M})} \le C_{\mathrm{err}}C_\rho^{q_{\mathrm{cert}}}\frac{\sqrt{d}}{L}; \qquad \|g\|_{L^2_{\mu\infty}(\mathcal{M})} \le C_{\mathrm{cert}}\rho_{\min}^{-q_{\mathrm{cert}}}\frac{\sqrt{d}}{n}$$

*and $\tau > 0$ is chosen such that*

$$\tau \le \frac{c'}{nL},$$

*then one has generalization in $L_{\mu\infty}^2(\mathcal{M})$:*

$$\mathbb{P}\left[\left\|\zeta_{\lfloor L^q/(n\tau)\rfloor}^N\right\|_{L_{\mu\infty}^2(\mathcal{M})} \leq \frac{C'C_\rho^{q_{\mathrm{cert}}}\sqrt{d}\log L}{L^q}\right] \geq 1 - \frac{C'''Le^{-cd}}{n\tau},$$

*and in addition, one has generalization in $L^\infty(\mathcal{M})$:*

$$\mathbb{P}\left[\left\|\zeta_{\lfloor L^q/(n\tau)\rfloor}^N\right\|_{L^\infty(\mathcal{M})} \leq \frac{C''C_\rho^{1+2q_{\mathrm{cert}}/3}(1+\rho_{\max})^{1/2}e^{14/(3\delta)}}{\min\left\{\mu^\infty(\mathcal{M}_+), \mu^\infty(\mathcal{M}_-)\right\}^{1/2}} \frac{d^{3/4}\log^{4/3}L}{L^{(4q-3)/6}}\right] \geq 1 - \frac{C'''Le^{-cd}}{n\tau}.$$

*The constant $C_\rho = \max\{\rho_{\min}, \rho_{\min}^{-1}\}$.*

*Proof.* The proof controls the $L^\infty$ norm of the error evaluated along the finite sample dynamics using an interpolation inequality for Lipschitz functions on an interval (Lemma B.14), which relates the $L^\infty$ norm to a certain combination of the predictor's Lipschitz constant and its $L_{\mu\infty}^2$ norm. We can control these two quantities at time zero using our measure concentration results; to control them for larger times $0 < k \leq L^q/(n\tau)$, we set up a system of coupled 'discrete integral equations' for the generalization error of the finite sample predictor and the Lipschitz constant of the finite sample predictor, and use the fact that $k\tau$ is not large to argue by induction that not much blow-up can occur. Along the way, we control the generalization error of the finite sample predictor by linking it to the generalization error of the nominal predictor as controlled in Lemma B.6; the residual that arises is shown to be small by applying Corollary B.11 and applying basic results from optimal transport theory adapted to our setting, encapsulated in Lemmas B.13 and B.16.

To begin, we will lay out the probabilistic bounds we will rely on for simplifications, so that the rest of the proof can proceed without interruption. We will want to satisfy

$$\tau < \frac{1}{\max\left\{\left\|\Theta_{\mu^N}\right\|_{L_{\mu^N}^2(\mathcal{M})\to L_{\mu^N}^2(\mathcal{M})}, \left\|\Theta_{\mu^\infty}\right\|_{L_{\mu\infty}^2(\mathcal{M})\to L_{\mu\infty}^2(\mathcal{M})}\right\}}, \tag{B.23}$$

following the notation of Lemma B.10. Using Jensen's inequality, the Schwarz inequality, and the triangle inequality, we have for $\star \in \{N, \infty\}$

$$\left\|\Theta_{\mu^\star}\right\|_{L_{\mu^\star}^2(\mathcal{M})\to L_{\mu^\star}^2(\mathcal{M})} = \sup_{\|g\|_{L_{\mu^\star}^2(\mathcal{M})}\leq 1}\left\|\int_{\mathcal{M}}\Theta(\boldsymbol{x}, \boldsymbol{x}')g(\boldsymbol{x}')\,\mathrm{d}\mu^\star(\boldsymbol{x}')\right\|_{L_{\mu^\star}^2(\mathcal{M})}$$

$$\leq \|g\|_{L_{\mu^\star}^1(\mathcal{M})}\sup_{(\boldsymbol{x},\boldsymbol{x}')\in\mathcal{M}\times\mathcal{M}}|\Theta(\boldsymbol{x}, \boldsymbol{x}')|$$

$$\leq \begin{matrix}\sup_{(\boldsymbol{x},\boldsymbol{x}')\in\mathcal{M}\times\mathcal{M}}|\Theta(\boldsymbol{x}, \boldsymbol{x}') - \psi_1\circ\angle(\boldsymbol{x}, \boldsymbol{x}')| \\ + \sup_{(\boldsymbol{x},\boldsymbol{x}')\in\mathcal{M}\times\mathcal{M}}|\psi_1\circ\angle(\boldsymbol{x}, \boldsymbol{x}')|,\end{matrix} \tag{B.24}$$

where the notation $\psi_1$ follows the definition in Appendix C.2.2. The first term in (B.24) can be controlled using Theorem B.2: we obtain that on an event of probability at least $1 - e^{-cd}$

$$\|\Theta - \psi_1\circ\angle\|_{L^\infty(\mathcal{M}\times\mathcal{M})} \leq \sqrt{d^4nL^3} \tag{B.25}$$

if $d \geq Kd_0\log(nn_0C_\mathcal{M})$ and $n \geq K'd^4L$. The second term in (B.24) can be controlled using the triangle inequality, Lemma E.5, and the definition of $\psi_1$: we obtain that it is no larger than $nL/2$. Combining these two bounds, we have on an event of probability at least $1 - e^{-cd}$

$$\max\left\{\left\|\Theta_{\mu^N}\right\|_{L_{\mu^N}^2(\mathcal{M})\to L_{\mu^N}^2(\mathcal{M})}, \left\|\Theta_{\mu^\infty}\right\|_{L_{\mu\infty}^2(\mathcal{M})\to L_{\mu\infty}^2(\mathcal{M})}\right\} \leq CnL \tag{B.26}$$

provided $d \geq Kd_0\log(nn_0C_\mathcal{M})$ and $n \geq K'd^4L$. Thus, with probability at least $1 - e^{-cd}$, our choice of step size $\tau \leq c/(nL)$ satisfies (B.23). Under our hypotheses on the function $g$ in the statement of the result and taking a union bound with the event in (B.26), we can invoke Lemma B.6 to obtain

$$\mathbb{P}\left[\bigcap_{C\sqrt{d}/(n\tau\rho_{\min}^{q_{\mathrm{cert}}})\leq k\leq L^q/(n\tau)}\left\{\|\zeta_k^\infty\|_{L_{\mu\infty}^2(\mathcal{M})} \leq \frac{C'C_\rho^{q_{\mathrm{cert}}}\sqrt{d}\log L}{nk\tau}\right\}\right] \geq 1 - \frac{C'''Le^{-cd}}{n\tau} \tag{B.27}$$

and

$$\mathbb{P}\left[\bigcap_{0 \le k \le L^q/(n\tau)} \left\{\sum_{s=0}^{k} \|\zeta_s^\infty\|_{L_{\mu^\infty}^2(\mathcal{M})} \le C_\rho^{2q_{\mathrm{cert}}} \frac{C''d\log^2 L}{n\tau}\right\}\right] \ge 1 - \frac{C'''Le^{-cd}}{n\tau} \qquad \text{(B.28)}$$

provided $d \ge Kd_0 \log(nn_0 C_\mathcal{M})$, $1/2 \le q < 1$, $n \ge K'd^4 L^5$, and $L \ge K''C_\rho^{2q_{\mathrm{cert}}}d$. We have by Lemmas B.6 and B.10, a union bound with (B.26), and our condition on $\tau$ that

$$\mathbb{P}\left[\bigcap_{0 \le k \le L^q/(n\tau)} \left\{\|\zeta_k^\infty\|_{L_{\mu^\infty}^2(\mathcal{M})} \le \sqrt{d}\right\} \cap \bigcap_{0 \le k \le L^q/(n\tau)} \left\{\|\zeta_k^N\|_{L_{\mu^N}^2(\mathcal{M})} \le \sqrt{d}\right\}\right] \ge 1 - \frac{CLe^{-cd}}{n\tau}$$
$$\text{(B.29)}$$

as long as $d \ge Kd_0 \log(nn_0 C_\mathcal{M})$ and $n \ge K'L^{48+20q}d^9 \log^9 L$, and where we used our conditions on $\tau$ and $q$ to obtain that $L^q/n\tau \ge 1$ and simplify the probability bound; and, following the notation of Corollary B.11, we have by this result (again under our condition on $\tau$ and a union bound) that there is an event of probability at least $1 - CLe^{-cd}/(n\tau)$ on which

$$\Delta_{\lfloor L^q/(n\tau)\rfloor - 1}^N \le \left(n^{11}L^{48+8q}d^9 \log^9 L\right)^{1/12} \qquad \text{(B.30)}$$

under the previous conditions on $n$ and $d$. In addition, applying Lemma D.12 and a union bound gives that on an event of probability at least $1 - Ce^{-cd}$

$$\max\left\{\left\|\zeta|_{\mathcal{M}_+}\right\|_{\mathrm{Lip}}, \left\|\zeta|_{\mathcal{M}_-}\right\|_{\mathrm{Lip}}\right\} \le \sqrt{d} \qquad \text{(B.31)}$$

provided $d \ge Kd_0 \log(nn_0 C_\mathcal{M})$ and $n \ge K' \max\{d^4 L, (\kappa/c_\lambda)^{1/3}, \kappa^{2/5}\}$. Finally, we have by Lemma B.13 that for any $0 < \delta \le 1$

$$\mathbb{P}\left[\bigcap_{f \in \mathrm{Lip}(\mathcal{M})} \left\{\begin{array}{c} \left|\int_\mathcal{M} f(\boldsymbol{x})\,\mathrm{d}\mu^\infty(\boldsymbol{x}) - \int_\mathcal{M} f(\boldsymbol{x})\,\mathrm{d}\mu^N(\boldsymbol{x})\right| \\ \le \frac{2\|f\|_{L^\infty(\mathcal{M})}\sqrt{d}}{N} + \frac{e^{14/\delta}C_{\mu^\infty,\mathcal{M}}\sqrt{d}\max_{\star\in\{+,-\}}\left\|f|_{\mathcal{M}_\star}\right\|_{\mathrm{Lip}}}{N^{1/(2+\delta)}} \end{array}\right\}\right] \ge 1 - 8e^{-d}, \text{ (B.32)}$$

as long as $d \ge 1$ and $N \ge 2\sqrt{d}/\min\{\mu^\infty(\mathcal{M}_+), \mu^\infty(\mathcal{M}_-)\}$. We let $\mathcal{E}(q,\delta)$ denote the event consisting of the union of the events appearing in the bounds (B.25) to (B.32) hold; by a union bound and the previous observation that $L^q/n\tau \ge 1$, we have

$$\mathbb{P}[\mathcal{E}] \ge 1 - \frac{C'Le^{-cd}}{n\tau}.$$

In the sequel, we will use the events defining $\mathcal{E}$ to simplify our residuals without explicitly referencing that our bounds hold only on $\mathcal{E}$ to save time.

We start from the dynamics update equations given by Lemma B.8, which we use to write

$$\zeta_k^\infty - \zeta_k^N = (\mathrm{Id} - \tau\boldsymbol{\Theta})\left[\zeta_{k-1}^\infty - \zeta_{k-1}^N\right] + \tau\boldsymbol{\Theta}_{k-1}^N\left[\zeta_{k-1}^N\right] - \tau\boldsymbol{\Theta}\left[\zeta_{k-1}^N\right],$$

where $\boldsymbol{\Theta}$ is defined as in Lemma B.12. Under the choice of $\tau$ and positivity of $\boldsymbol{\Theta}$ (Lemma B.9), we apply the triangle inequality and a telescoping series with the common initial conditions to obtain

$$\left\|\zeta_k^\infty - \zeta_k^N\right\|_{L_{\mu^\infty}^2(\mathcal{M})} \le \tau \sum_{s=0}^{k-1} \left\|\boldsymbol{\Theta}_s^N\left[\zeta_s^N\right] - \boldsymbol{\Theta}\left[\zeta_s^N\right]\right\|_{L_{\mu^\infty}^2(\mathcal{M})}. \qquad \text{(B.33)}$$

We can write

$$\boldsymbol{\Theta}_s^N\left[\zeta_s^N\right](\boldsymbol{x}) = \int_\mathcal{M} \Theta_s^N(\boldsymbol{x},\boldsymbol{x}')\zeta_s^N(\boldsymbol{x})\,\mathrm{d}\mu^N(\boldsymbol{x}')$$
$$= \int_\mathcal{M} \left(\Theta_s^N(\boldsymbol{x},\boldsymbol{x}') - \psi_1 \circ \angle(\boldsymbol{x},\boldsymbol{x}')\right)\zeta_s^N(\boldsymbol{x}')\,\mathrm{d}\mu^N(\boldsymbol{x}')$$
$$+ \int_\mathcal{M} \psi_1 \circ \angle(\boldsymbol{x},\boldsymbol{x}')\zeta_s^N(\boldsymbol{x}')\,\mathrm{d}\mu^N(\boldsymbol{x}'),$$

and analogously

$$\boldsymbol{\Theta}\left[\zeta_s^N\right](\boldsymbol{x}) = \int_\mathcal{M} \left(\Theta(\boldsymbol{x},\boldsymbol{x}') - \psi_1 \circ \angle(\boldsymbol{x},\boldsymbol{x}')\right)\zeta_s^N(\boldsymbol{x}')\,\mathrm{d}\mu^\infty(\boldsymbol{x}')$$
$$+ \int_\mathcal{M} \psi_1 \circ \angle(\boldsymbol{x},\boldsymbol{x}')\zeta_s^N(\boldsymbol{x}')\,\mathrm{d}\mu^\infty(\boldsymbol{x}').$$

Using Jensen's inequality and the Schwarz inequality, we have

$$
\left\| \int_{\mathcal{M}} \left( \Theta_s^N(\boldsymbol{x}, \boldsymbol{x}') - \psi_1 \circ \angle(\boldsymbol{x}, \boldsymbol{x}') \right) \zeta_s^N(\boldsymbol{x}') \, \mathrm{d}\mu^N(\boldsymbol{x}') \right\|_{L_{\mu^\infty}^2(\mathcal{M})}
$$
$$
\leq \int_{\mathcal{M}} \left\| \Theta_s^N(\,\cdot\,, \boldsymbol{x}') - \psi_1 \circ \angle(\,\cdot\,, \boldsymbol{x}') \right\|_{L_{\mu^\infty}^2(\mathcal{M})} |\zeta_s^N(\boldsymbol{x}')| \, \mathrm{d}\mu^N(\boldsymbol{x}')
$$
$$
\leq \left\| \Theta_s^N - \psi_1 \circ \angle \right\|_{L^\infty(\mathcal{M}\times\mathcal{M})} \left\| \zeta_s^N \right\|_{L_{\mu^N}^1(\mathcal{M})}
$$
$$
\leq \left\| \Theta_s^N - \psi_1 \circ \angle \right\|_{L^\infty(\mathcal{M}\times\mathcal{M})} \left\| \zeta_s^N \right\|_{L_{\mu^N}^2(\mathcal{M})},
$$

since $\mu^N$ is a probability measure. Repeating an analogous calculation with $\mu^\infty$ for the other term and applying the triangle inequality, we have

$$
\left\| \boldsymbol{\Theta}_s^N \left[ \zeta_s^N \right] - \boldsymbol{\Theta} \left[ \zeta_s^N \right] \right\|_{L_{\mu^\infty}^2(\mathcal{M})} \leq \left\| \boldsymbol{\Theta} - \psi_1 \circ \angle \right\|_{L^\infty(\mathcal{M}\times\mathcal{M})} \left( \begin{array}{c} \left\| \zeta_s^\infty - \zeta_s^N \right\|_{L_{\mu^\infty}^2(\mathcal{M})} \\ + \left\| \zeta_s^\infty \right\|_{L_{\mu^\infty}^2(\mathcal{M})} \\ + \left\| \zeta_s^N \right\|_{L_{\mu^N}^2(\mathcal{M})} \end{array} \right)
$$
$$
+ \left\| \boldsymbol{\Theta}_s^N - \boldsymbol{\Theta} \right\|_{L^\infty(\mathcal{M}\times\mathcal{M})} \left\| \zeta_s^N \right\|_{L_{\mu^N}^2(\mathcal{M})}
$$
$$
+ \left\| \int_{\mathcal{M}} \psi_1 \circ \angle(\,\cdot\,, \boldsymbol{x}') \zeta_s^N(\boldsymbol{x}') \left( \mathrm{d}\mu^\infty(\boldsymbol{x}') - \mathrm{d}\mu^N(\boldsymbol{x}') \right) \right\|_{L_{\mu^\infty}^2(\mathcal{M})}.
$$
$$
\tag{B.34}
$$

We detour briefly to simplify residuals appearing in (B.34) before using the result to update (B.33). Using (B.25) and (B.30), we get

$$
\left\| \boldsymbol{\Theta} - \psi_1 \circ \angle \right\|_{L^\infty(\mathcal{M}\times\mathcal{M})} \left( \left\| \zeta_s^\infty - \zeta_s^N \right\|_{L_{\mu^\infty}^2(\mathcal{M})} + \left\| \zeta_s^\infty \right\|_{L_{\mu^\infty}^2(\mathcal{M})} + \left\| \zeta_s^N \right\|_{L_{\mu^N}^2(\mathcal{M})} \right)
$$
$$
+ \left\| \boldsymbol{\Theta}_s^N - \boldsymbol{\Theta} \right\|_{L^\infty(\mathcal{M}\times\mathcal{M})} \left\| \zeta_s^N \right\|_{L_{\mu^N}^2(\mathcal{M})}
$$
$$
\leq \begin{array}{c} \sqrt{d^4 n L^3} \left( \left\| \zeta_s^\infty - \zeta_s^N \right\|_{L_{\mu^\infty}^2(\mathcal{M})} + \left\| \zeta_s^\infty \right\|_{L_{\mu^\infty}^2(\mathcal{M})} + \left\| \zeta_s^N \right\|_{L_{\mu^N}^2(\mathcal{M})} \right) \\ + \left( n^{11} L^{48+8q} d^9 \log^9 L \right)^{1/12} \left\| \zeta_s^N \right\|_{L_{\mu^N}^2(\mathcal{M})} \end{array}
$$
$$
\leq \left( n^{11} L^{48+8q} d^9 \log^9 L \right)^{1/12} \left( \left\| \zeta_s^\infty - \zeta_s^N \right\|_{L_{\mu^\infty}^2(\mathcal{M})} + \left\| \zeta_s^\infty \right\|_{L_{\mu^\infty}^2(\mathcal{M})} + 2 \left\| \zeta_s^N \right\|_{L_{\mu^N}^2(\mathcal{M})} \right).
$$
$$
\tag{B.35}
$$

where the final bound holds when $n \geq d^3$. Using (B.29), we can further simplify the RHS of the last bound above to

$$
\left( n^{11} L^{48+8q} d^9 \log^9 L \right)^{1/12} \left( \left\| \zeta_s^\infty - \zeta_s^N \right\|_{L_{\mu^\infty}^2(\mathcal{M})} + \left\| \zeta_s^\infty \right\|_{L_{\mu^\infty}^2(\mathcal{M})} + 2 \left\| \zeta_s^N \right\|_{L_{\mu^N}^2(\mathcal{M})} \right)
$$
$$
\leq 2 \left( n^{11} L^{48+8q} d^{15} \log^9 L \right)^{1/12} + \left( n^{11} L^{48+8q} d^9 \log^9 L \right)^{1/12} \left\| \zeta_s^\infty - \zeta_s^N \right\|_{L_{\mu^\infty}^2(\mathcal{M})}.
$$

With this last bound and (B.34), we can use $k\tau \leq L^q/n$ to simplify (B.33) to

$$
\left\| \zeta_k^\infty - \zeta_k^N \right\|_{L_{\mu^\infty}^2(\mathcal{M})} \leq \begin{array}{c} C \left( \dfrac{L^{48+20} d^{15} \log^9 L}{n} \right)^{1/12} \\ + \tau \left( n^{11} L^{48+8q} d^9 \log^9 L \right)^{1/12} \sum_{s=0}^{k-1} \left\| \zeta_s^\infty - \zeta_s^N \right\|_{L_{\mu^\infty}^2(\mathcal{M})} \\ + \tau \sum_{s=0}^{k-1} \left\| \int_{\mathcal{M}} \psi_1 \circ \angle(\,\cdot\,, \boldsymbol{x}') \zeta_s^N(\boldsymbol{x}') \left( \mathrm{d}\mu^\infty(\boldsymbol{x}') - \mathrm{d}\mu^N(\boldsymbol{x}') \right) \right\|_{L_{\mu^\infty}^2(\mathcal{M})}. \end{array}
$$
$$
\tag{B.36}
$$

To control the remaining term in (B.36), we split the error $\zeta_s^N$ into a Lipschitz component whose evolution is governed by the nominal kernel $\psi_1 \circ \angle$ and a nonsmooth component which is small in

$L^\infty$. Formally, we define $\Theta^{\mathrm{nom}} : L^2_{\mu^N}(\mathcal{M}) \to L^2_{\mu^N}(\mathcal{M})$ by

$$\Theta^{\mathrm{nom}} [g] (\boldsymbol{x}) = \int_{\mathcal{M}} \psi_1 \circ \angle(\boldsymbol{x}, \boldsymbol{x}') g(\boldsymbol{x}') \, \mathrm{d}\mu^N(\boldsymbol{x}'),$$

and use the update equation from Lemma B.8 to write

$$\zeta^N_s = \zeta - \tau \sum_{i=0}^{s-1} \Theta^N_i \left[\zeta^N_i\right]$$

$$= \underbrace{\zeta - \tau \sum_{i=0}^{s-1} \Theta^{\mathrm{nom}} \left[\zeta^N_i\right]}_{\zeta^{N,\mathrm{Lip}}_s} + \underbrace{\tau \sum_{i=0}^{s-1} \left(\Theta^{\mathrm{nom}} - \Theta^N_i\right) \left[\zeta^N_i\right]}_{\delta^N_s},$$

so that $\zeta^N_s = \zeta^{N,\mathrm{Lip}}_s + \delta^N_s$, and $\zeta^{N,\mathrm{Lip}}_0 = \zeta$, $\delta^N_0 = 0$. It is straightforward to control $\delta^N_s$ in $L^\infty$: we have (as usual) by the triangle inequality, Jensen's inequality, and the Schwarz inequality

$$\left\|\delta^N_s\right\|_{L^\infty(\mathcal{M})} \leq \tau \sum_{i=0}^{s-1} \int_{\mathcal{M}} \left\|\psi_1 \circ \angle(\,\cdot\,, \boldsymbol{x}') - \Theta^N_i(\,\cdot\,, \boldsymbol{x}')\right\|_{L^\infty(\mathcal{M})} \left|\zeta^N_i(\boldsymbol{x}')\right| \mathrm{d}\mu^N(\boldsymbol{x}')$$

$$\leq \tau \sum_{i=0}^{s-1} \left\|\psi_1 \circ \angle - \Theta^N_i\right\|_{L^\infty(\mathcal{M}\times\mathcal{M})} \left\|\zeta^N_i\right\|_{L^2_{\mu^N}(\mathcal{M})},$$

and then the triangle inequality together with (B.25), (B.29) and (B.30) yield

$$\left\|\delta^N_s\right\|_{L^\infty(\mathcal{M})} \leq s\tau\sqrt{d} \left(\sqrt{d^4 n L^3} + \left(n^{11} L^{48+8q} d^9 \log^9 L\right)^{1/12}\right)$$

$$\leq s\tau\sqrt{d} \left(n^{11} L^{48+8q} d^9 \log^9 L\right)^{1/12}, \tag{B.37}$$

where the second line applies the same simplifications that led us to (B.35). The triangle inequality gives

$$\left\|\int_{\mathcal{M}} \psi_1 \circ \angle(\,\cdot\,, \boldsymbol{x}') \delta^N_s(\boldsymbol{x}') \left(\mathrm{d}\mu^\infty(\boldsymbol{x}') - \mathrm{d}\mu^N(\boldsymbol{x}')\right)\right\|_{L^2_{\mu^\infty}(\mathcal{M})}$$

$$\leq \sum_{\star \in \{N,\infty\}} \left\|\int_{\mathcal{M}} \psi_1 \circ \angle(\,\cdot\,, \boldsymbol{x}') \delta^N_s(\boldsymbol{x}') \, \mathrm{d}\mu^\star(\boldsymbol{x}')\right\|_{L^2_{\mu^\infty}(\mathcal{M})},$$

and simplifying as usual using Jensen's inequality and the Hölder inequality, we obtain

$$\left\|\int_{\mathcal{M}} \psi_1 \circ \angle(\,\cdot\,, \boldsymbol{x}') \delta^N_s(\boldsymbol{x}') \left(\mathrm{d}\mu^\infty(\boldsymbol{x}') - \mathrm{d}\mu^N(\boldsymbol{x}')\right)\right\|_{L^2_{\mu^\infty}(\mathcal{M})} \leq nL \left\|\delta^N_s\right\|_{L^\infty(\mathcal{M})}$$

$$\leq s\tau \left(n^{23} L^{60+8q} d^{15} \log^9 L\right)^{1/12},$$

where the last bound uses (B.37). Then using the triangle inequality and $k\tau \leq L^q/n$ to simplify in (B.36), we obtain

$$\left\|\zeta^\infty_k - \zeta^N_k\right\|_{L^2_{\mu^\infty}(\mathcal{M})} \leq C \left(\frac{L^{60+32q} d^{15} \log^9 L}{n}\right)^{1/12}$$

$$+ \tau \left(n^{11} L^{48+8q} d^9 \log^9 L\right)^{1/12} \sum_{s=0}^{k-1} \left\|\zeta^\infty_s - \zeta^N_s\right\|_{L^2_{\mu^\infty}(\mathcal{M})}$$

$$+ \tau \sum_{s=0}^{k-1} \left\|\int_{\mathcal{M}} \psi_1 \circ \angle(\,\cdot\,, \boldsymbol{x}') \zeta^{N,\mathrm{Lip}}_s(\boldsymbol{x}') \left(\mathrm{d}\mu^\infty(\boldsymbol{x}') - \mathrm{d}\mu^N(\boldsymbol{x}')\right)\right\|_{L^2_{\mu^\infty}(\mathcal{M})}. \tag{B.38}$$

To simplify the remaining term in (B.38), we aim to apply (B.32); to do this we will need to justify the notation and establish that $\zeta^{N,\mathrm{Lip}}_s \in \mathrm{Lip}(\mathcal{M})$ regardless of the random sample from $\mu^\infty$ and the

random instance of the weights. Because $\zeta_s^{N,\mathrm{Lip}}$ is a sum of functions, we can bound its minimal Lipschitz constant by the sum of bounds on the Lipschitz constants of each summand. We always have for either $\star \in \{+, -\}$

$$\left\| \zeta_s^{N,\mathrm{Lip}} \big|_{\mathcal{M}_\star} \right\|_{\mathrm{Lip}} \le \left\| \zeta \big|_{\mathcal{M}_\star} \right\|_{\mathrm{Lip}} + \tau \sum_{i=0}^{s-1} \left\| \int_{\mathcal{M}} \psi_1 \circ \angle(\,\cdot\,, \boldsymbol{x}') \zeta_i^N(\boldsymbol{x}') \, \mathrm{d}\mu^N(\boldsymbol{x}') \right\|_{\mathrm{Lip}}. \qquad (\mathrm{B}.39)$$

We note that because the ReLU $[\,\cdot\,]_+$ is 1-Lipschitz as a map on $\mathbb{R}^n$, we have

$$\left\| \zeta \big|_{\mathcal{M}_\star} \right\|_{\mathrm{Lip}} \le \| \boldsymbol{W}^{L+1} \|_2 \prod_{\ell=1}^{L} \| \boldsymbol{W}^\ell \| < +\infty,$$

so we need only develop a Lipschitz property for the summands in the second term of (B.39). To do this, we will start by showing that $t \mapsto \psi_1 \circ \cos^{-1}\langle \boldsymbol{\gamma}_\star(t), \boldsymbol{x}' \rangle$ is absolutely continuous for each $\boldsymbol{x}'$. Continuity is immediate. The only obstruction to differentiability comes from the inverse cosine, which fails to be differentiable at $\pm 1$, and because $\mathcal{M} \subset \mathbb{S}^{n_0-1}$ we have $\langle \boldsymbol{\gamma}_\star(t), \boldsymbol{x}' \rangle = \pm 1$ only if $\boldsymbol{\gamma}_\star(t) = \pm \boldsymbol{x}'$; because $\boldsymbol{\gamma}_\star$ are simple curves, this shows that there are at most two points of nondifferentiability in $[0, \mathrm{len}(\mathcal{M}_\star)]$. At points of differentiability, we calculate using the chain rule the derivative

$$t \mapsto - \left( \psi_1' \circ \cos^{-1}\langle \boldsymbol{\gamma}_\star(t), \boldsymbol{x}' \rangle \right) \left\langle \frac{\boldsymbol{\gamma}_\star'(t)}{\sqrt{1 - \langle \boldsymbol{\gamma}_\star(t), \boldsymbol{x}' \rangle^2}}, \boldsymbol{x}' \right\rangle,$$

and because $\boldsymbol{\gamma}_\star$ is a sphere curve, it holds $(\boldsymbol{I} - \boldsymbol{\gamma}_\star(t)\boldsymbol{\gamma}_\star^*(t))\boldsymbol{\gamma}_\star'(t) = \boldsymbol{\gamma}_\star'(t)$ for all $t$, whence by Cauchy-Schwarz

$$\left| \left\langle \frac{\boldsymbol{\gamma}_\star'(t)}{\sqrt{1 - \langle \boldsymbol{\gamma}_\star(t), \boldsymbol{x}' \rangle^2}}, \boldsymbol{x}' \right\rangle \right| = \left| \left\langle \frac{(\boldsymbol{I} - \boldsymbol{\gamma}_\star(t)\boldsymbol{\gamma}_\star(t)^*)\boldsymbol{x}'}{\sqrt{1 - \langle \boldsymbol{\gamma}_\star(t), \boldsymbol{x}' \rangle^2}}, \boldsymbol{\gamma}_\star'(t) \right\rangle \right|$$
$$\le \frac{\| (\boldsymbol{I} - \boldsymbol{\gamma}_\star(t)\boldsymbol{\gamma}_\star(t)^*)\boldsymbol{x}' \|_2}{\sqrt{1 - \langle \boldsymbol{\gamma}_\star(t), \boldsymbol{x}' \rangle^2}} \le 1, \qquad (\mathrm{B}.40)$$

where we also used that $\boldsymbol{\gamma}_\star$ are unit-speed curves. In particular, the derivative is bounded, hence integrable on $[0, \mathrm{len}(\mathcal{M}_\star)]$, and so an application of (Cohn, 2013, Theorem 6.3.11) establishes that $t \mapsto \psi_1 \circ \cos^{-1}\langle \boldsymbol{\gamma}_\star(t), \boldsymbol{x}' \rangle$ is absolutely continuous, with the expansion

$$\left| \psi_1 \circ \cos^{-1}\langle \boldsymbol{\gamma}_\star(t), \boldsymbol{x}' \rangle - \psi_1 \circ \cos^{-1}\langle \boldsymbol{\gamma}_\star(t'), \boldsymbol{x}' \rangle \right|$$
$$= \left| \int_t^{t'} \left( \psi_1' \circ \cos^{-1}\langle \boldsymbol{\gamma}_\star(t''), \boldsymbol{x}' \rangle \right) \left\langle \frac{\boldsymbol{\gamma}_\star'(t'')}{\sqrt{1 - \langle \boldsymbol{\gamma}_\star(t''), \boldsymbol{x}' \rangle^2}}, \boldsymbol{x}' \right\rangle \mathrm{d}t'' \right|,$$

which gives an avenue to establish Lipschitz estimates for $t \mapsto \psi_1 \circ \cos^{-1}\langle \boldsymbol{\gamma}_\star(t), \boldsymbol{x}' \rangle$. Because $\boldsymbol{x}' \mapsto \zeta_i^N(\boldsymbol{x}')$ is continuous and $i \le s \le k \le L^q/(n\tau) < +\infty$, an application of Fubini's theorem enables us to also use this result to obtain Lipschitz estimates for the summands examined in (B.39), to wit

$$\left\| \int_{\mathcal{M}} \psi_1 \circ \angle(\,\cdot\,, \boldsymbol{x}') \zeta_i^N(\boldsymbol{x}') \, \mathrm{d}\mu^N(\boldsymbol{x}') \right\|_{\mathrm{Lip}}$$
$$\le \sup_{\boldsymbol{x} \in \mathcal{M}_\star} \int_{\mathcal{M}} |\psi_1' \circ \angle(\boldsymbol{x}, \boldsymbol{x}')| |\zeta_i^N(\boldsymbol{x}')| \, \mathrm{d}\mu^N(\boldsymbol{x}')$$
$$\le \| \zeta_i^N \|_{L_{\mu^N}^2(\mathcal{M})} \sup_{\boldsymbol{x} \in \mathcal{M}_\star} \left( \int_{\mathcal{M}} \left( \psi_1' \circ \angle(\boldsymbol{x}, \boldsymbol{x}') \right)^2 \mathrm{d}\mu^N(\boldsymbol{x}') \right)^{1/2} \qquad (\mathrm{B}.41)$$

after using the bound (B.40) in the first inequality and the Schwarz inequality for the second. Before proceeding with further simplifications, we note that the $C^2$ property of $\psi_1$, continuity of $\zeta_i^N$, boundedness of $i$, and compactness of $\mathcal{M}$ let us assert using (B.41) and (B.39) that $\zeta_s^{N,\mathrm{Lip}} \in \mathrm{Lip}(\mathcal{M})$ whether or not we are working on the event $\mathcal{E}$. Continuing, we develop a bound for the RHS of (B.41) that is valid on $\mathcal{E}$. Using the triangle inequality and the Minkowski inequality, we have for

the second term on the RHS of the last bound in (B.41)

$$
\sup_{\boldsymbol{x}\in\mathcal{M}_\star} \left( \int_{\mathcal{M}} (\psi_1' \circ \angle(\boldsymbol{x}, \boldsymbol{x}'))^2 \, \mathrm{d}\mu^N(\boldsymbol{x}') \right)^{1/2}
$$

$$
\leq \sup_{\boldsymbol{x}\in\mathcal{M}_\star} \left( \left| \int_{\mathcal{M}} (\psi_1' \circ \angle(\boldsymbol{x}, \boldsymbol{x}'))^2 \left( \mathrm{d}\mu^N(\boldsymbol{x}') - \mathrm{d}\mu^\infty(\boldsymbol{x}') \right) \right| \right)^{1/2} \tag{B.42}
$$

$$
+ \sup_{\boldsymbol{x}\in\mathcal{M}_\star} \left( \int_{\mathcal{M}} (\psi_1' \circ \angle(\boldsymbol{x}, \boldsymbol{x}'))^2 \, \mathrm{d}\mu^\infty(\boldsymbol{x}') \right)^{1/2}.
$$

For the first term in (B.42), we use Lemmas C.7, C.22 and C.24 to obtain that $\boldsymbol{x}' \mapsto (\psi_1' \circ \angle(\boldsymbol{x}, \boldsymbol{x}'))^2$ is bounded by $Cn^2L^4$ and $C'n^2L^5$-Lipschitz for every $\boldsymbol{x}$, and then applying (B.32) gives

$$
\sup_{\boldsymbol{x}\in\mathcal{M}_\star} \left( \left| \int_{\mathcal{M}} (\psi_1' \circ \angle(\boldsymbol{x}, \boldsymbol{x}'))^2 \left( \mathrm{d}\mu^N(\boldsymbol{x}') - \mathrm{d}\mu^\infty(\boldsymbol{x}') \right) \right| \right)^{1/2}
$$

$$
\leq \left( \frac{Cn^2L^4\sqrt{d}}{N} + \frac{e^{14/\delta}C_{\mu^\infty,\mathcal{M}}C'n^2L^5\sqrt{d}}{N^{1/(2+\delta)}} \right)^{1/2}
$$

$$
\leq C \frac{(1 + C_{\mu^\infty,\mathcal{M}})^{1/2}e^{7/\delta}}{N^{1/(4+2\delta)}} nL^{5/2}d^{1/4}. \tag{B.43}
$$

For the second term in (B.42), we apply Lemmas C.8 and C.22 together with the choice $L \geq K\kappa^2 C_\lambda$ to get

$$
\sup_{\boldsymbol{x}\in\mathcal{M}_\star} \left( \int_{\mathcal{M}} (\psi_1' \circ \angle(\boldsymbol{x}, \boldsymbol{x}'))^2 \, \mathrm{d}\mu^\infty(\boldsymbol{x}') \right)^{1/2} \leq CnL^2 \sup_{\boldsymbol{x}\in\mathcal{M}_\pm} \left( \int_{\mathcal{M}} \frac{\mathrm{d}\mu^\infty(\boldsymbol{x}')}{(1 + (L/\pi)\angle(\boldsymbol{x}, \boldsymbol{x}'))^2} \right)^{1/2}
$$

$$
\leq CnL^{3/2}\rho_{\max}^{1/2}(\mathrm{len}(\mathcal{M}_+) + \mathrm{len}(\mathcal{M}_-))^{1/2}
$$

$$
\leq C\rho_{\max}^{1/2}C_{\mu^\infty,\mathcal{M}}^{1/2}nL^{3/2}. \tag{B.44}
$$

Combining (B.43) and (B.44) to control the RHS of (B.42), we obtain from (B.41)

$$
\left\| \int_{\mathcal{M}} \psi_1 \circ \angle(\,\cdot\,, \boldsymbol{x}')\zeta_i^N(\boldsymbol{x}') \, \mathrm{d}\mu^N(\boldsymbol{x}') \right\|_{\mathrm{Lip}}
$$

$$
\leq C\|\zeta_i^N\|_{L^2_{\mu^N}(\mathcal{M})} \left( \frac{(1 + C_{\mu^\infty,\mathcal{M}})^{1/2}e^{7/\delta}}{N^{1/(4+2\delta)}} nL^{5/2}d^{1/4} + \rho_{\max}C_{\mu^\infty,\mathcal{M}}nL^{3/2} \right)
$$

$$
\leq C\|\zeta_i^N\|_{L^2_{\mu^N}(\mathcal{M})} (1 + C_{\mu^\infty,\mathcal{M}})^{1/2} e^{7/\delta}(1 + \rho_{\max})^{1/2}d^{1/4}nL^{3/2}, \tag{B.45}
$$

where in the second line we used $N \geq L^{4+2\delta}$. Plugging (B.45) into (B.39) and applying in addition (B.31), we get

$$
\left\| \zeta_s^{N,\mathrm{Lip}} \big|_{\mathcal{M}_\star} \right\|_{\mathrm{Lip}} \leq \sqrt{d} + C\tau e^{7/\delta} (1 + C_{\mu^\infty,\mathcal{M}})^{1/2} (1 + \rho_{\max})^{1/2}d^{1/4}nL^{3/2} \sum_{i=0}^{s-1} \|\zeta_i^N\|_{L^2_{\mu^N}(\mathcal{M})}. \tag{B.46}
$$

Let us briefly pause to reorient ourselves. We do not have control of the empirical losses appearing in (B.46) by an outside result, so we need to make some further simplifications to this bound. We will control the sum of empirical losses term in (B.46) by linking it to the difference population error, which we last saw in (B.38), and the population error using the triangle inequality and a change of measure inequality. Meanwhile, with the Lipschitz property of $\zeta_s^{N,\mathrm{Lip}}$ we have shown, we will be able to obtain a bound in terms of simpler quantities for the last term on the RHS of (B.38) using (B.32). The two resulting bounds will give us a system of two coupled 'discrete integral equations' for the difference population error and the Lipschitz constants of $\zeta_s^{N,\mathrm{Lip}}$, which we will solve inductively.

First, we continue simplifying (B.46). The triangle inequality and the fact that $\mu^N$ is a probability measure give

$$
\|\zeta_i^N\|_{L^2_{\mu^N}(\mathcal{M})} \leq \left\| \zeta_i^{N,\mathrm{Lip}} \right\|_{L^2_{\mu^N}(\mathcal{M})} + \|\delta_i^N\|_{L^\infty(\mathcal{M})}, \tag{B.47}
$$

and we have by the triangle inequality and Hölder-$\frac{1}{2}$ continuity of $x \mapsto \sqrt{x}$

$$
\left\| \zeta_i^{N,\text{Lip}} \right\|_{L^2_{\mu^N}(\mathcal{M})} \leq \left\| \zeta_i^{N,\text{Lip}} \right\|_{L^2_{\mu^\infty}(\mathcal{M})} + \left| \left\| \zeta_i^{N,\text{Lip}} \right\|_{L^2_{\mu^N}(\mathcal{M})} - \left\| \zeta_i^{N,\text{Lip}} \right\|_{L^2_{\mu^\infty}(\mathcal{M})} \right|
$$

$$
\leq \left\| \zeta_i^{N,\text{Lip}} \right\|_{L^2_{\mu^\infty}(\mathcal{M})} + \sqrt{\left| \int_{\mathcal{M}} \left( \zeta_i^{N,\text{Lip}}(\boldsymbol{x}) \right)^2 (\mathrm{d}\mu^\infty(\boldsymbol{x}) - \mathrm{d}\mu^N(\boldsymbol{x})) \right|}. \quad \text{(B.48)}
$$

We have shown that $\zeta_i^{N,\text{Lip}} \in \text{Lip}(\mathcal{M})$ and $\zeta_i^{N,\text{Lip}} \in L^\infty(\mathcal{M})$ above, and so $\left( \zeta_i^{N,\text{Lip}} \right)^2 \in \text{Lip}(\mathcal{M})$ as well, with

$$
\left\| \left( \zeta_i^{N,\text{Lip}} \right)^2 \Big|_{\mathcal{M}_\star} \right\|_{\text{Lip}} \leq 2 \left\| \zeta_i^{N,\text{Lip}} \right\|_{L^\infty(\mathcal{M})} \left\| \zeta_i^{N,\text{Lip}} \Big|_{\mathcal{M}_\star} \right\|_{\text{Lip}}.
$$

Applying the previous equation with (B.32) to control (B.48), we get

$$
\left\| \zeta_i^{N,\text{Lip}} \right\|_{L^2_{\mu^N}(\mathcal{M})}
$$

$$
\leq \quad \begin{aligned} & \left\| \zeta_i^{N,\text{Lip}} \right\|_{L^2_{\mu^\infty}(\mathcal{M})} \\ & + Cd^{1/4} \sqrt{ \frac{\left\| \zeta_i^{N,\text{Lip}} \right\|_{L^\infty(\mathcal{M})}^2}{N} + \frac{e^{14/\delta} C_{\mu^\infty,\mathcal{M}} \left\| \zeta_i^{N,\text{Lip}} \right\|_{L^\infty(\mathcal{M})} \max_{\star \in \{+,-\}} \left\| \zeta_i^{N,\text{Lip}} \Big|_{\mathcal{M}_\star} \right\|_{\text{Lip}}}{N^{1/(2+\delta)}} } \end{aligned}
$$

$$
\leq \quad \begin{aligned} & \left\| \zeta_i^{N,\text{Lip}} \right\|_{L^2_{\mu^\infty}(\mathcal{M})} \\ & + Cd^{1/4} \left( \frac{\left\| \zeta_i^{N,\text{Lip}} \right\|_{L^\infty(\mathcal{M})}}{\sqrt{N}} + \frac{e^{7/\delta} C_{\mu^\infty,\mathcal{M}}^{1/2} \left\| \zeta_i^{N,\text{Lip}} \right\|_{L^\infty(\mathcal{M})}^{1/2} \max_{\star \in \{+,-\}} \left\| \zeta_i^{N,\text{Lip}} \Big|_{\mathcal{M}_\star} \right\|_{\text{Lip}}^{1/2}}{N^{1/(4+2\delta)}} \right), \end{aligned}
$$

where the second line applies the Minkowski inequality. Using the triangle inequality and that $\mu^\infty$ is a probability measure, we have

$$
\left\| \zeta_i^{N,\text{Lip}} \right\|_{L^2_{\mu^\infty}(\mathcal{M})} \leq \left\| \zeta_i^N \right\|_{L^2_{\mu^\infty}(\mathcal{M})} + \left\| \delta_i^N \right\|_{L^2_{\mu^\infty}(\mathcal{M})}
$$

$$
\leq \left\| \zeta_i^\infty \right\|_{L^2_{\mu^\infty}(\mathcal{M})} + \left\| \zeta_i^N - \zeta_i^\infty \right\|_{L^2_{\mu^\infty}(\mathcal{M})} + \left\| \delta_i^N \right\|_{L^\infty(\mathcal{M})}. \quad \text{(B.49)}
$$

Substituting (B.49) into (B.47) and using (B.37) to simplify gives

$$
\left\| \zeta_i^N \right\|_{L^2_{\mu^N}(\mathcal{M})}
$$

$$
\leq \left\| \zeta_i^N - \zeta_i^\infty \right\|_{L^2_{\mu^\infty}(\mathcal{M})} + \left\| \zeta_i^\infty \right\|_{L^2_{\mu^\infty}(\mathcal{M})} + 2i\tau\sqrt{d} \left( n^{11} L^{48+8q} d^9 \log^9 L \right)^{1/12}
$$

$$
+ Cd^{1/4} \left( \frac{\left\| \zeta_i^{N,\text{Lip}} \right\|_{L^\infty(\mathcal{M})}}{\sqrt{N}} + \frac{e^{7/\delta} C_{\mu^\infty,\mathcal{M}}^{1/2} \left\| \zeta_i^{N,\text{Lip}} \right\|_{L^\infty(\mathcal{M})}^{1/2} \max_{\star \in \{+,-\}} \left\| \zeta_i^{N,\text{Lip}} \Big|_{\mathcal{M}_\star} \right\|_{\text{Lip}}^{1/2}}{N^{1/(4+2\delta)}} \right).
$$

$$
\text{(B.50)}
$$

Following (B.46), we need to sum the previous bound over $i$. To simplify residuals, we use (B.28) to get

$$
Cs^2 \tau \sqrt{d} \left( n^{11} L^{48+8q} d^9 \log^9 L \right)^{1/12} + \sum_{i=0}^{s-1} \left\| \zeta_i^\infty \right\|_{L^2_{\mu^\infty}(\mathcal{M})}
$$

$$
\leq Cs^2 \tau \sqrt{d} \left( n^{11} L^{48+8q} d^9 \log^9 L \right)^{1/12} + \frac{C_\rho^{2q_{\text{cert}}} C' d \log^2 L}{n\tau}
$$

$$
\leq \frac{2 C_\rho^{2q_{\text{cert}}} C' d \log^2 L}{n\tau},
$$

where the second bound uses the control $s\tau \le k\tau \le L^q/n$ and holds under the condition $n \ge (C/C')^{12}L^{48+32q}d^3$. Summing in (B.50) and using the previous bound, it follows

$$
\begin{aligned}
&\sum_{i=0}^{s-1}\big\|\zeta_i^N\big\|_{L^2_{\mu^N}(\mathcal{M})} \\
&\le \frac{CC_\rho^{2q_{\mathrm{cert}}}d\log^2 L}{n\tau} + \sum_{i=0}^{s-1}\big\|\zeta_i^N - \zeta_i^\infty\big\|_{L^2_{\mu^\infty}(\mathcal{M})} \\
&\quad + Cd^{1/4}\left(\frac{\big\|\zeta_i^{N,\mathrm{Lip}}\big\|_{L^\infty(\mathcal{M})}}{\sqrt{N}} + \frac{e^{7/\delta}C_{\mu^\infty,\mathcal{M}}^{1/2}\big\|\zeta_i^{N,\mathrm{Lip}}\big\|_{L^\infty(\mathcal{M})}^{1/2}\max_{\star\in\{+,-\}}\big\|\zeta_i^{N,\mathrm{Lip}}\big|_{\mathcal{M}_\star}\big\|_{\mathrm{Lip}}^{1/2}}{N^{1/(4+2\delta)}}\right).
\end{aligned}
$$

(B.51)

Plugging (B.51) into (B.46), we obtain

$$
\begin{aligned}
&\Big\|\zeta_s^{N,\mathrm{Lip}}\big|_{\mathcal{M}_\star}\Big\|_{\mathrm{Lip}} \\
&\le C_1 d^{1/4}L^{3/2}\left(\begin{array}{c} d\log^2 L + n\tau\sum_{i=0}^{s-1}\big\|\zeta_i^N - \zeta_i^\infty\big\|_{L^2_{\mu^\infty}(\mathcal{M})} \\[4pt] +n\tau d^{1/4}\sum_{i=0}^{s-1}\frac{\big\|\zeta_i^{N,\mathrm{Lip}}\big\|_{L^\infty(\mathcal{M})}}{\sqrt{N}} + \frac{\big\|\zeta_i^{N,\mathrm{Lip}}\big\|_{L^\infty(\mathcal{M})}^{1/2}\max_{\star\in\{+,-\}}\big\|\zeta_i^{N,\mathrm{Lip}}\big|_{\mathcal{M}_\star}\big\|_{\mathrm{Lip}}^{1/2}}{N^{1/(4+2\delta)}} \end{array}\right),
\end{aligned}
$$

(B.52)

where for concision we have defined

$$
C_1(\delta, \mu^\infty) = CC_\rho^{2q_{\mathrm{cert}}}e^{14/\delta}\left(1 + C_{\mu^\infty,\mathcal{M}}\right)\left(1 + \rho_{\max}\right)^{1/2}
$$

(B.53)

and simplified the $\sqrt{d}$ residual in (B.46) by worst-casing with the larger residual from the population error term in (B.51), and made other simplifications by worst-casing some constants. We simplify (B.38) next: we have shown that $\zeta_s^{N,\mathrm{Lip}} \in \mathrm{Lip}(\mathcal{M})$ and $\zeta_s^{N,\mathrm{Lip}} \in L^\infty(\mathcal{M})$ above, and so for every $\boldsymbol{x} \in \mathcal{M}$, we have

$$
\psi_1 \circ \angle(\boldsymbol{x}, \cdot)\zeta_s^{N,\mathrm{Lip}} \in \mathrm{Lip}(\mathcal{M})
$$

as well, with

$$
\Big\|\psi_1 \circ \angle(\boldsymbol{x}, \cdot)\zeta_s^{N,\mathrm{Lip}}\big|_{\mathcal{M}_\star}\Big\|_{\mathrm{Lip}} \le CnL\max_{\star'\in\{+,-\}}\Big\|\zeta_s^{N,\mathrm{Lip}}\big|_{\mathcal{M}_{\star'}}\Big\|_{\mathrm{Lip}} + C'nL^2\big\|\zeta_s^{N,\mathrm{Lip}}\big\|_{L^\infty(\mathcal{M})}
$$

(B.54)

using the definition of $\psi_1$, Lemmas E.5, C.7 and C.22, and

$$
\big\|\psi_1 \circ \angle(\boldsymbol{x}, \cdot)\zeta_s^{N,\mathrm{Lip}}\big\|_{L^\infty(\mathcal{M})} \le CnL\big\|\zeta_s^{N,\mathrm{Lip}}\big\|_{L^\infty(\mathcal{M})}.
$$

(B.55)

The bounds (B.54) and (B.55) retain no $\boldsymbol{x}$ dependence. Applying (B.32) and integrating over $\boldsymbol{x}$, we obtain from (B.54) and (B.55)

$$
\begin{aligned}
&\left\|\int_{\mathcal{M}}\psi_1 \circ \angle(\cdot, \boldsymbol{x}')\zeta_s^{N,\mathrm{Lip}}(\boldsymbol{x}')\left(\mathrm{d}\mu^\infty(\boldsymbol{x}') - \mathrm{d}\mu^N(\boldsymbol{x}')\right)\right\|_{L^2_{\mu^\infty}(\mathcal{M})} \\
&\le \frac{CnL\sqrt{d}\big\|\zeta_s^{N,\mathrm{Lip}}\big\|_{L^\infty(\mathcal{M})}}{N} \\
&\quad + \frac{CnLe^{14/\delta}C_{\mu^\infty,\mathcal{M}}\sqrt{d}\max_{\star\in\{+,-\}}\Big\|\zeta_s^{N,\mathrm{Lip}}\big|_{\mathcal{M}_\star}\Big\|_{\mathrm{Lip}}}{N^{1/(2+\delta)}} \\
&\quad + \frac{CnL^2e^{14/\delta}C_{\mu^\infty,\mathcal{M}}\sqrt{d}\big\|\zeta_s^{N,\mathrm{Lip}}\big\|_{L^\infty(\mathcal{M})}}{N^{1/(2+\delta)}},
\end{aligned}
$$

and we can combine the first and third terms on the RHS of the previous bound by worst-casing, giving

$$\left\| \int_{\mathcal{M}} \psi_1 \circ \angle(\,\cdot\,, \boldsymbol{x}') \zeta_s^{N,\mathrm{Lip}}(\boldsymbol{x}') \left( \mathrm{d}\mu^\infty(\boldsymbol{x}') - \mathrm{d}\mu^N(\boldsymbol{x}') \right) \right\|_{L^2_{\mu\infty}(\mathcal{M})}$$

$$\leq \frac{C\sqrt{d}nLe^{14/\delta}\left(1 + C_{\mu^\infty,\mathcal{M}}\right)}{N^{1/(2+\delta)}} \left( \max_{\star \in \{+,-\}} \left\| \zeta_s^{N,\mathrm{Lip}} \big|_{\mathcal{M}_\star} \right\|_{\mathrm{Lip}} + L \left\| \zeta_s^{N,\mathrm{Lip}} \right\|_{L^\infty(\mathcal{M})} \right).$$

Plugging the previous bound into (B.38), we obtain

$$\left\| \zeta_k^\infty - \zeta_k^N \right\|_{L^2_{\mu\infty}(\mathcal{M})}$$

$$\leq C \left( \frac{L^{60+32q} d^{15} \log^9 L}{n} \right)^{1/12} + \tau \left( n^{11} L^{48+8q} d^9 \log^9 L \right)^{1/12} \sum_{s=0}^{k-1} \left\| \zeta_s^\infty - \zeta_s^N \right\|_{L^2_{\mu\infty}(\mathcal{M})}$$

$$+ \frac{C\tau\sqrt{d}nLe^{14/\delta}\left(1 + C_{\mu^\infty,\mathcal{M}}\right)}{N^{1/(2+\delta)}} \sum_{s=0}^{k-1} \left( \max_{\star \in \{+,-\}} \left\| \zeta_s^{N,\mathrm{Lip}} \big|_{\mathcal{M}_\star} \right\|_{\mathrm{Lip}} + L \left\| \zeta_s^{N,\mathrm{Lip}} \right\|_{L^\infty(\mathcal{M})} \right).$$
(B.56)

To finish coupling (B.52) and (B.56), we need to remove the $L^\infty(\mathcal{M})$ terms. We accomplish this using Lemma B.14, which gives

$$\left\| \zeta_s^{N,\mathrm{Lip}} \right\|_{L^\infty(\mathcal{M})} \leq C C_2^{1/2} \left\| \zeta_s^{N,\mathrm{Lip}} \right\|_{L^2_{\mu\infty}(\mathcal{M})} + \frac{C}{\rho_{\min}^{1/3}} \left\| \zeta_s^{N,\mathrm{Lip}} \right\|_{L^2_{\mu\infty}(\mathcal{M})}^{2/3} \max_{\star \in \{+,-\}} \left\| \zeta_s^{N,\mathrm{Lip}} \big|_{\mathcal{M}_\star} \right\|_{\mathrm{Lip}}^{1/3},$$
(B.57)

where we have defined

$$C_2(\mu^\infty) = \frac{\rho_{\max}}{\rho_{\min} \min \{\mu^\infty(\mathcal{M}_+), \mu^\infty(\mathcal{M}_-)\}}.$$
(B.58)

For coupling purposes, it will suffice to use a version of (B.57) obtained by simplifying with some coarse estimates. Using (B.49), (B.37) and (B.29), we have

$$\left\| \zeta_i^{N,\mathrm{Lip}} \right\|_{L^2_{\mu\infty}(\mathcal{M})} \leq \sqrt{d} + i\tau\sqrt{d} \left( n^{11} L^{48+8q} d^9 \log^9 L \right)^{1/12} + \left\| \zeta_i^N - \zeta_i^\infty \right\|_{L^2_{\mu\infty}(\mathcal{M})}$$

$$\leq 2\sqrt{d} + \left\| \zeta_i^N - \zeta_i^\infty \right\|_{L^2_{\mu\infty}(\mathcal{M})},$$

using $i\tau \leq L^q/n$ and $n \geq L^{48+20q} d^9 \log^9 L$ in the second line, and plugging this into (B.57) and using the Minkowski inequality gives

$$\left\| \zeta_s^{N,\mathrm{Lip}} \right\|_{L^\infty(\mathcal{M})} \leq C C_2^{1/2} \sqrt{d} + C C_2^{1/2} \left\| \zeta_i^N - \zeta_i^\infty \right\|_{L^2_{\mu\infty}(\mathcal{M})} + \frac{C d^{1/3}}{\rho_{\min}^{1/3}} \max_{\star \in \{+,-\}} \left\| \zeta_s^{N,\mathrm{Lip}} \big|_{\mathcal{M}_\star} \right\|_{\mathrm{Lip}}^{1/3}$$

$$+ \frac{C}{\rho_{\min}^{1/3}} \left\| \zeta_i^N - \zeta_i^\infty \right\|_{L^2_{\mu\infty}(\mathcal{M})}^{2/3} \max_{\star \in \{+,-\}} \left\| \zeta_s^{N,\mathrm{Lip}} \big|_{\mathcal{M}_\star} \right\|_{\mathrm{Lip}}^{1/3}.$$
(B.59)

To make some of the subsequent bounds more concise, we introduce additional notation

$$\Lambda_s = \max_{\star \in \{+,-\}} \left\| \zeta_s^{N,\mathrm{Lip}} \big|_{\mathcal{M}_\star} \right\|_{\mathrm{Lip}}.$$

Plugging (B.59) into (B.52) and using the Minkowski inequality, we obtain

$$
\begin{aligned}
\Lambda_s \leq CC_1 d^{1/4} L^{3/2} &\left( d \log^2 L + \frac{C_2^{1/2} d^{3/4} n s \tau}{\sqrt{N}} + n\tau \left( 1 + \frac{C_2^{1/2} d^{1/4}}{\sqrt{N}} \right) \sum_{i=0}^{s-1} \big\| \zeta_i^N - \zeta_i^\infty \big\|_{L_\mu^{2\infty}(\mathcal{M})} \right. \\
&+ \frac{n\tau d^{7/12}}{\rho_{\min}^{1/3} \sqrt{N}} \sum_{i=0}^{s-1} \Lambda_i^{1/3} + \frac{C_2^{1/4} n\tau d^{1/2}}{N^{1/(4+2\delta)}} \sum_{i=0}^{s-1} \Lambda_i^{1/2} \\
&+ \frac{n\tau d^{5/12}}{\rho_{\min}^{1/6} N^{1/(4+2\delta)}} \sum_{i=0}^{s-1} \Lambda_i^{2/3} + \frac{n\tau d^{1/4}}{\rho_{\min}^{1/3} \sqrt{N}} \sum_{i=0}^{s-1} \big\| \zeta_i^N - \zeta_i^\infty \big\|_{L_\mu^{2\infty}(\mathcal{M})}^{2/3} \Lambda_i^{1/3} \\
&+ \frac{C_2^{1/4} n\tau d^{1/4}}{N^{1/(4+2\delta)}} \sum_{i=0}^{s-1} \big\| \zeta_i^N - \zeta_i^\infty \big\|_{L_\mu^{2\infty}(\mathcal{M})}^{1/2} \Lambda_i^{1/2} \\
&\left. + \frac{n\tau d^{1/4}}{\rho_{\min}^{1/6} N^{1/(4+2\delta)}} \sum_{i=0}^{s-1} \big\| \zeta_i^N - \zeta_i^\infty \big\|_{L_\mu^{2\infty}(\mathcal{M})}^{1/3} \Lambda_i^{2/3} \right).
\end{aligned}
\tag{B.60}
$$

To simplify (B.60), we use $s\tau \leq L^q/n$, $C_2 \geq 1$, and $q \leq 1$, and so if additionally we choose $N \geq C_2 \max\{\sqrt{d}, L^2\}$ we obtain

$$
\begin{aligned}
\Lambda_s \leq CC_1 d^{1/4} L^{3/2} &\left( d \log^2 L + n\tau \sum_{i=0}^{s-1} \big\| \zeta_i^N - \zeta_i^\infty \big\|_{L_\mu^{2\infty}(\mathcal{M})} \right. \\
&+ \frac{n\tau d^{1/4}}{\rho_{\min}^{1/6} N^{1/(4+2\delta)}} \sum_{i=0}^{s-1} \big\| \zeta_i^N - \zeta_i^\infty \big\|_{L_\mu^{2\infty}(\mathcal{M})}^{1/3} \Lambda_i^{2/3} + \frac{n\tau d^{7/12}}{\rho_{\min}^{1/3} \sqrt{N}} \sum_{i=0}^{s-1} \Lambda_i^{1/3} \\
&+ \frac{C_2^{1/4} n\tau d^{1/2}}{N^{1/(4+2\delta)}} \sum_{i=0}^{s-1} \Lambda_i^{1/2} + \frac{n\tau d^{5/12}}{\rho_{\min}^{1/6} N^{1/(4+2\delta)}} \sum_{i=0}^{s-1} \Lambda_i^{2/3} \\
&+ \frac{n\tau d^{1/4}}{\rho_{\min}^{1/3} \sqrt{N}} \sum_{i=0}^{s-1} \big\| \zeta_i^N - \zeta_i^\infty \big\|_{L_\mu^{2\infty}(\mathcal{M})}^{2/3} \Lambda_i^{1/3} \\
&\left. + \frac{C_2^{1/4} n\tau d^{1/4}}{N^{1/(4+2\delta)}} \sum_{i=0}^{s-1} \big\| \zeta_i^N - \zeta_i^\infty \big\|_{L_\mu^{2\infty}(\mathcal{M})}^{1/2} \Lambda_i^{1/2} \right).
\end{aligned}
\tag{B.61}
$$

Meanwhile, we recall

$$
C_{\mu^\infty, \mathcal{M}} = \frac{\text{len}(\mathcal{M}_+)}{\mu^\infty(\mathcal{M}_+)} + \frac{\text{len}(\mathcal{M}_-)}{\mu^\infty(\mathcal{M}_-)},
$$

and an integration in coordinates gives

$$
\mu^\infty(\mathcal{M}_\pm) = \int_0^{\text{len}(\mathcal{M}_\pm)} \rho_\pm \circ \gamma_\pm(t) \, dt \geq \rho_{\min} \, \text{len}(\mathcal{M}_\pm),
$$

so that

$$
C_{\mu^\infty, \mathcal{M}} \leq \frac{2}{\rho_{\min}}.
\tag{B.62}
$$

Using (B.62) and plugging (B.59) into (B.56), we obtain

$$\left\| \zeta_k^\infty - \zeta_k^N \right\|_{L_\mu^{2\infty}(\mathcal{M})}$$

$$\leq C \left( \frac{L^{60+32q} d^{15} \log^9 L}{n} \right)^{1/12}$$

$$+ \tau \left( \left( n^{11} L^{48+8q} d^9 \log^9 L \right)^{1/12} + \frac{C' C_2^{1/2} \sqrt{d} n L^2 e^{14/\delta}}{\rho_{\min} N^{1/(2+\delta)}} \right) \sum_{s=0}^{k-1} \left\| \zeta_s^\infty - \zeta_s^N \right\|_{L_\mu^{2\infty}(\mathcal{M})}$$

$$+ \frac{C' \tau \sqrt{d} n L e^{14/\delta}}{\rho_{\min} N^{1/(2+\delta)}} \sum_{s=0}^{k-1} \left( \begin{array}{c} \Lambda_s + L C_2^{1/2} \sqrt{d} \\ + L d^{1/3} \rho_{\min}^{-1/3} \Lambda_s^{1/3} + L \rho_{\min}^{-1/3} \left\| \zeta_s^\infty - \zeta_s^N \right\|_{L_\mu^{2\infty}(\mathcal{M})}^{2/3} \Lambda_s^{1/3} \end{array} \right) . \tag{B.63}$$

In (B.61) and (B.63), we now have a suitable system of coupled discrete integral equations for $\left\| \zeta_k^\infty - \zeta_k^N \right\|_{L_\mu^{2\infty}(\mathcal{M})}$ and $\Lambda_k$. We will solve these equations by positing bounds for each parameter that are valid for all indices $0 \leq k \leq \lfloor L^q/(n\tau) \rfloor$ based on inspection of (B.61) and (B.63), then proving the bounds hold by induction on $k$. Positing the bounds is not too hard, because each term in (B.61) and (B.63) with a factor of $N$ in its denominator can be forced to be small by requiring $N$ to be large enough. For all $0 \leq k \leq \lfloor L^q/(n\tau) \rfloor$, we claim

$$\left\| \zeta_k^\infty - \zeta_k^N \right\|_{L_\mu^{2\infty}(\mathcal{M})} \leq C_{\text{diff}} \max \left\{ \begin{array}{c} C_{\text{lip}} C_1 C_2^{1/2} C_\rho^{4/3} \frac{d^{7/4} L^{5/2+q} \log^2 L}{N^{1/(2+\delta)}}, \\ \left( \frac{L^{60+32q} d^{15} \log^9 L}{n} \right)^{1/12} \end{array} \right\} \tag{B.64}$$

$$\Lambda_k \leq C_{\text{lip}} C_1 d^{5/4} L^{3/2} \log^2 L, \tag{B.65}$$

where $C_{\text{diff}}$ and $C_{\text{lip}}$ are two absolute constants that we will specify in our arguments below. We prove (B.64) and (B.65) by induction on $k$. The case of $k = 0$ is immediate, since $\zeta_0^\infty = \zeta_0^N$ for (B.64); and by construction $\zeta_0^{N,\text{Lip}} = \zeta$, and (B.31) and $d \geq 1$ then gives (B.65) if $L \geq e$. We therefore move to the induction step, assuming that (B.64) and (B.65) hold for $k - 1$ and showing that this implies the bounds for $k$. We begin by verifying (B.64). Applying the induction hypothesis for $k - 1$ via (B.65), we can write

$$\Lambda_s + L C_2^{1/2} \sqrt{d} + L \left( \frac{d \Lambda_s}{\rho_{\min}} \right)^{1/3}$$

$$\leq C_{\text{lip}} C_1 d^{5/4} L^{3/2} \log^2 L + L C_2^{1/2} \sqrt{d} + \left( \frac{C_{\text{lip}} C_1}{\rho_{\min}} \right)^{1/3} d^{3/4} L^{3/2} \log^{2/3} L$$

$$\leq C_{\text{lip}} C_1 C_2^{1/2} C_\rho^{1/3} d^{5/4} L^{3/2} \log^2 L,$$

where we worst-cased in the second line using $C_{\text{lip}} \geq 1$ and $C_1 \geq 1$, $C_2 \geq 1$, which follow from (B.53) and (B.58). We use $k\tau \leq L^q/n$ with the last bound to note that

$$\frac{C' \tau \sqrt{d} n L e^{14/\delta}}{\rho_{\min} N^{1/(2+\delta)}} \sum_{s=0}^{k-1} C_{\text{lip}} C_1 C_2^{1/2} C_\rho^{1/3} d^{5/4} L^{3/2} \log^2 L \leq C'' C_{\text{lip}} C_1 C_2^{1/2} C_\rho^{4/3} \frac{d^{7/4} L^{5/2+q} \log^2 L}{N^{1/(2+\delta)}},$$

where $C'' \geq 1$. Using this bound and (B.65) once more, we can simplify (B.63) to

$$\left\| \zeta_k^\infty - \zeta_k^N \right\|_{L_\mu^{2\infty}(\mathcal{M})}$$

$$\leq C'' C_{\text{lip}} C_1 C_2^{1/2} C_\rho^{4/3} \frac{d^{7/4} L^{5/2+q} \log^2 L}{N^{1/(2+\delta)}} + C \left( \frac{L^{60+32q} d^{15} \log^9 L}{n} \right)^{1/12}$$

$$+ \tau \left( \left( n^{11} L^{48+8q} d^9 \log^9 L \right)^{1/12} + \frac{C' C_2^{1/2} \sqrt{d} n L^2 e^{14/\delta}}{\rho_{\min} N^{1/(2+\delta)}} \right) \sum_{s=0}^{k-1} \left\| \zeta_s^\infty - \zeta_s^N \right\|_{L_\mu^{2\infty}(\mathcal{M})} \tag{B.66}$$

$$+ \frac{C' C_{\text{lip}}^{1/3} C_1^{1/3} e^{14/\delta}}{\rho_{\min}^{4/3}} \frac{\tau d^{11/12} n L^{5/2} \log^{2/3} L}{N^{1/(2+\delta)}} \sum_{s=0}^{k-1} \left\| \zeta_s^\infty - \zeta_s^N \right\|_{L_\mu^{2\infty}(\mathcal{M})}^{2/3} .$$

Noticing that the RHS of the bound (B.64) does not depend on $k$, let us momentarily denote it by $C_{\mathrm{diff}} M$ (i.e., the part of the RHS of this bound that does not involve $C_{\mathrm{diff}}$ is denoted as $M$). Plugging into (B.66) and using $k\tau \leq L^q/n$, we obtain

$$\left\|\zeta_k^\infty - \zeta_k^N\right\|_{L^2_{\mu\infty}(\mathcal{M})} \leq C'' C_{\mathrm{lip}} C_1 C_2^{1/2} C_\rho^{4/3} \frac{d^{7/4} L^{5/2+q} \log^2 L}{N^{1/(2+\delta)}} + C\left(\frac{L^{60+32q} d^{15} \log^9 L}{n}\right)^{1/12}$$

$$+ C_{\mathrm{diff}}\left(\left(\frac{L^{48+20q} d^9 \log^9 L}{n}\right)^{1/12} + \frac{C' C_2^{1/2}\sqrt{d} L^{2+q} e^{14/\delta}}{\rho_{\min} N^{1/(2+\delta)}}\right) M$$

$$+ \frac{C' C_{\mathrm{diff}}^{2/3} C_{\mathrm{lip}}^{1/3} C_1^{1/3} e^{14/\delta}}{\rho_{\min}^{4/3}} \frac{d^{11/12} L^{5/2+q} \log^{2/3} L}{N^{1/(2+\delta)}} M^{2/3}.$$

In particular, if $C_{\mathrm{diff}} = 6\max\{C, C''\}$ (for the constants in the first line of the previous bound), we can bound the RHS of the previous bound and obtain

$$\left\|\zeta_k^\infty - \zeta_k^N\right\|_{L^2_{\mu\infty}(\mathcal{M})} \leq \frac{C_{\mathrm{diff}} M}{3} + C_{\mathrm{diff}}\left(\left(\frac{L^{48+20q} d^9 \log^9 L}{n}\right)^{1/12} + \frac{C' C_2^{1/2}\sqrt{d} L^{2+q} e^{14/\delta}}{\rho_{\min} N^{1/(2+\delta)}}\right) M$$

$$+ \frac{C' C_{\mathrm{diff}}^{2/3} C_{\mathrm{lip}}^{1/3} C_1^{1/3} e^{14/\delta}}{\rho_{\min}^{4/3}} \frac{d^{11/12} L^{5/2+q} \log^{2/3} L}{N^{1/(2+\delta)}} M^{2/3}.$$

$$\text{(B.67)}$$

We can conclude (B.64) from (B.67) provided we can show the second and third terms are no larger than $C_{\mathrm{diff}} M/3$. For the second term in (B.67), if we choose $N$ such that

$$N^{1/(2+\delta)} \geq 6C' C_2^{1/2} \rho_{\min}^{-1} e^{14/\delta} d^{1/2} L^{2+q}$$

and $n$ such that

$$n \geq 6^{12} L^{48+20q} d^9 \log^9 L$$

then we have

$$C_{\mathrm{diff}}\left(\left(\frac{L^{48+20q} d^9 \log^9 L}{n}\right)^{1/12} + \frac{C' C_2^{1/2}\sqrt{d} L^{2+q} e^{14/\delta}}{\rho_{\min} N^{1/(2+\delta)}}\right) M \leq \frac{C_{\mathrm{diff}} M}{3}.$$

For the third term in (B.67), we proceed in cases: first, when

$$C_{\mathrm{lip}} C_1 C_2^{1/2} C_\rho^{4/3} \frac{d^{7/4} L^{5/2+q} \log^2 L}{N^{1/(2+\delta)}} \leq \left(\frac{L^{60+32q} d^{15} \log^9 L}{n}\right)^{1/12}, \qquad \text{(B.68)}$$

we have by (B.64)

$$M = \left(\frac{L^{60+32q} d^{15} \log^9 L}{n}\right)^{1/12},$$

and if we require additionally $C_{\mathrm{diff}} \geq 1$, it follows that

$$\frac{C' C_{\mathrm{diff}}^{2/3} C_{\mathrm{lip}}^{1/3} C_1^{1/3} e^{14/\delta}}{\rho_{\min}^{4/3}} \frac{d^{11/12} L^{5/2+q} \log^{2/3} L}{N^{1/(2+\delta)}} M^{2/3}$$

$$\leq C' C_{\mathrm{diff}} C_{\mathrm{lip}} C_1 C_2^{1/2} C_\rho^{4/3} e^{14/\delta} \frac{d^{7/4} L^{5/2+q} \log^2 L}{N^{1/(2+\delta)}} M^{2/3}$$

$$\leq C' C_{\mathrm{diff}} e^{14/\delta} M^{1+2/3},$$

using $C_1 \geq 1$, $C_2 \geq 1$, and $C_{\mathrm{lip}} \geq 1$ and worst-casing exponents on $d$ and $\log L$ in the first line, and (B.68) in the second line. In particular, by the value of $M$ in this regime, if

$$n \geq (3C' e^{14/\delta})^{18} L^{60+32q} d^{15} \log^9 L$$

then we obtain for the third term in (B.67)

$$\frac{C' C_{\mathrm{diff}}^{2/3} C_{\mathrm{lip}}^{1/3} C_1^{1/3} e^{14/\delta}}{\rho_{\min}^{4/3}} \frac{d^{11/12} L^{5/2+q} \log^{2/3} L}{N^{1/(2+\delta)}} M^{2/3} \leq \frac{C_{\mathrm{diff}} M}{3},$$

as desired. Next, we consider the remaining case

$$C_{\text{lip}}C_1C_2^{1/2}C_\rho^{4/3}\frac{d^{7/4}L^{5/2+q}\log^2 L}{N^{1/(2+\delta)}} \geq \left(\frac{L^{60+32q}d^{15}\log^9 L}{n}\right)^{1/12}, \tag{B.69}$$

which by (B.64) implies

$$M = C_{\text{lip}}C_1C_2^{1/2}C_\rho^{4/3}\frac{d^{7/4}L^{5/2+q}\log^2 L}{N^{1/(2+\delta)}}.$$

With this setting of $M$, the third term in (B.67) can be bounded as

$$\frac{C'C_{\text{diff}}^{2/3}C_{\text{lip}}^{1/3}C_1^{1/3}e^{14/\delta}}{\rho_{\min}^{4/3}}\frac{d^{11/12}L^{5/2+q}\log^{2/3}L}{N^{1/(2+\delta)}}M^{2/3}$$

$$= C'C_{\text{diff}}^{2/3}C_{\text{lip}}C_1C_2^{1/3}C_\rho^{4/3+8/9}e^{14/\delta}\frac{d^{7/4+1/3}L^{5/2+q}L^{5/3+2q/3}\log^2 L}{N^{1/(2+\delta)+2/(6+3\delta)}}$$

$$\leq C'C_{\text{diff}}C_\rho^{8/9}e^{14/\delta}\frac{d^{1/3}L^{5/3+2q/3}}{N^{2/(6+3\delta)}}M,$$

and using the RHS of the final bound in the previous expression, we see that if we choose

$$N^{1/(2+\delta)} \geq (3C')^{3/2}C_\rho^{4/3}e^{21/\delta}d^{1/2}L^{5/2+q},$$

then we have for the case (B.69)

$$\frac{C'C_{\text{diff}}^{2/3}C_{\text{lip}}^{1/3}C_1^{1/3}e^{14/\delta}}{\rho_{\min}^{4/3}}\frac{d^{11/12}L^{5/2+q}\log^{2/3}L}{N^{1/(2+\delta)}}M^{2/3} \leq \frac{C_{\text{diff}}M}{3}.$$

Combining the bounds on the third term in (B.67) over both cases (B.68) and (B.69), we have shown

$$\left\|\zeta_k^\infty - \zeta_k^N\right\|_{L^2_{\mu^\infty}(\mathcal{M})} \leq C_{\text{diff}}M,$$

which proves (B.64). Next, to verify (B.65), we proceed with a similar idea: the bound claimed in (B.65) corresponds to a constant multiple of the first term in parentheses in (B.61), so to establish (B.65) it suffices to show that each of the other terms in (B.61) is no larger than a certain constant. To work with the maximum operation in (B.64), we will again split the analysis into two cases. First, we consider the case where (B.69) holds, so that the maximum in (B.64) is achieved by the second argument. Plugging (B.64) and (B.65) into (B.61) and using $k\tau \leq L^q/n$, we get

$$\Lambda_k \leq CC_1d^{1/4}L^{3/2}\left(d\log^2 L + C_{\text{diff}}C_{\text{lip}}C_1C_2^{1/2}C_\rho^{4/3}\frac{d^{7/4}L^{5/2+2q}\log^2 L}{N^{1/(2+\delta)}}\right.$$

$$+ C_{\text{diff}}^{1/3}C_{\text{lip}}C_\rho^{11/18}C_1C_2^{1/6}\frac{d^{5/3}L^{11/6+4q/3}\log^2 L}{N^{6/(10+5\delta)}}$$

$$+ C_{\text{lip}}^{1/3}C_1^{1/3}C_\rho^{1/3}\frac{dL^{1/2+q}\log^{2/3}L}{\sqrt{N}} + C_{\text{lip}}^{1/2}C_1^{1/2}C_2^{1/4}\frac{d^{9/8}L^{3/4+q}\log L}{N^{1/(4+2\delta)}}$$

$$+ C_{\text{lip}}^{2/3}C_1^{2/3}C_\rho^{1/6}\frac{d^{5/4}L^{1+q}\log^{4/3}L}{N^{1/(4+2\delta)}}$$

$$+ C_{\text{diff}}^{2/3}C_{\text{lip}}C_1C_2^{1/3}C_\rho^{5/3}\frac{d^{11/6}L^{13/6+5q/3}\log^2 L}{N^{(5+\delta)/(4+2\delta)}}$$

$$\left.+ C_{\text{diff}}^{1/2}C_{\text{lip}}C_1C_2^{1/2}C_\rho^{2/3}\frac{d^{7/4}L^{2+3q/2}\log^2 L}{N^{1/(2+\delta)}}\right).$$

Using $C_{\text{lip}} \geq 1$, $C_1 \geq 1$, $C_\rho \geq 1$, and $C_2 \geq 1$, we can worst-case constants in the previous expression to simplify. We can then do some selective worst-casing of the exponents on $d$, $N$, and $L$ in all except the first term: we have evidently (to combine the first and last terms)

$$\frac{d^{7/4}L^{2+3q/2}\log^2 L}{N^{1/(2+\delta)}} \leq \frac{d^{7/4}L^{5/2+2q}\log^2 L}{N^{1/(2+\delta)}}$$

and (to combine the first and second terms)

$$\frac{d^{5/3}L^{11/6+4q/3}\log^2 L}{N^{6/(10+5\delta)}} \le \frac{d^{7/4}L^{5/2+2q}\log^2 L}{N^{1/(2+\delta)}},$$

and because $0 < \delta \le 1$, we have $1/(2+\delta) \le 1/2$ and $(5+\delta)/(4+2\delta) \ge 1$, and if $N \ge d^{1/12}$ this implies (to combine the first and second-to-last terms)

$$\frac{d^{11/6}L^{13/6+5q/3}\log^2 L}{N^{(5+\delta)/(4+2\delta)}} \le \frac{d^{7/4}L^{5/2+2q}\log^2 L}{N^{1/(2+\delta)}}.$$

We can worst-case the remaining three terms, and we thus obtain

$$\Lambda_k \le CC_1 d^{1/4}L^{3/2}\left(\begin{array}{c} d\log^2 L + 4C_{\text{diff}}C_{\text{lip}}C_1 C_2^{1/2}C_\rho^{5/3}\frac{d^{1+3/4}L^{5/2+2q}\log^2 L}{N^{1/(2+\delta)}} \\ +3C_{\text{lip}}^{2/3}C_1^{2/3}C_2^{1/4}C_\rho^{1/3}\frac{d^{1+1/4}L^{1+q}\log^{4/3} L}{N^{1/(4+2\delta)}} \end{array}\right).$$

We can then pick $C_{\text{lip}} = 3C$, and if

$$N^{1/(4+2\delta)} \ge 3(3C)^{2/3}C_1^{2/3}C_2^{1/4}C_\rho^{1/3}d^{1/4}L^{1+q},$$

and

$$N^{1/(2+\delta)} \ge 12CC_{\text{diff}}C_1 C_2^{1/2}C_\rho^{5/3}d^{3/4}L^{5/2+2q},$$

then it follows from the previous bound

$$\Lambda_k \le 3CC_1 d^{5/4}L^{3/2}\log^2 L,$$

which establishes (B.65) in the first case, where (B.69) holds. Next, we consider the remaining case where (B.68) holds, so that the maximum in (B.64) is saturated by the first argument. We start by grouping some terms in (B.61) so that it will be slightly easier to simplify later: we can write

$$\begin{aligned}
\Lambda_k \le CC_1 d^{1/4}L^{3/2}\bigg( & d\log^2 L + n\tau\sum_{s=0}^{k-1}\left\|\zeta_s^N - \zeta_s^\infty\right\|_{L_\mu^{2\infty}(\mathcal{M})} + \\
& \frac{n\tau d^{1/4}}{\rho_{\min}^{1/6}N^{1/(4+2\delta)}}\sum_{s=0}^{k-1}\left(\left\|\zeta_s^N - \zeta_s^\infty\right\|_{L_\mu^{2\infty}(\mathcal{M})}^{1/3} + d^{1/6}\right)\Lambda_s^{2/3} \\
& + \frac{n\tau d^{1/4}}{\rho_{\min}^{1/3}\sqrt{N}}\sum_{s=0}^{k-1}\left(\left\|\zeta_s^N - \zeta_s^\infty\right\|_{L_\mu^{2\infty}(\mathcal{M})}^{2/3} + d^{1/3}\right)\Lambda_s^{1/3} \\
& + \frac{C_2^{1/4}n\tau d^{1/4}}{N^{1/(4+2\delta)}}\sum_{s=0}^{k-1}\left(\left\|\zeta_s^N - \zeta_s^\infty\right\|_{L_\mu^{2\infty}(\mathcal{M})}^{1/2} + d^{1/4}\right)\Lambda_s^{1/2}\bigg).
\end{aligned}$$
(B.70)

By the case-defining condition (B.68) and (B.64), enforcing

$$n \ge C_{\text{diff}}^{12}L^{60+32q}d^9\log^9 L$$

implies

$$\left\|\zeta_s^N - \zeta_s^\infty\right\|_{L_\mu^{2\infty}(\mathcal{M})} + d^{1/2} \le 2d^{1/2},$$

and we can use this to simplify (B.70), obtaining

$$\begin{aligned}
\Lambda_k \le CC_1 d^{1/4}L^{3/2}\bigg( & d\log^2 L + n\tau\sum_{s=0}^{k-1}\left\|\zeta_s^N - \zeta_s^\infty\right\|_{L_\mu^{2\infty}(\mathcal{M})} + \frac{2n\tau d^{5/12}}{\rho_{\min}^{1/6}N^{1/(4+2\delta)}}\sum_{s=0}^{k-1}\Lambda_s^{2/3} \\
& + \frac{2n\tau d^{7/12}}{\rho_{\min}^{1/3}\sqrt{N}}\sum_{s=0}^{k-1}\Lambda_s^{1/3} + \frac{2C_2^{1/4}n\tau d^{1/2}}{N^{1/(4+2\delta)}}\sum_{s=0}^{k-1}\Lambda_s^{1/2}\bigg).
\end{aligned}$$
(B.71)

Plugging (B.64) and (B.65) into (B.71) and using $k\tau \leq L^q/n$ and (B.68), we get the bound

$$
\Lambda_k \leq CC_1 d^{1/4} L^{3/2} \Bigg( d\log^2 L + C_{\text{diff}} \left( \frac{L^{60+44q}d^{15}\log^9 L}{n} \right)^{1/12} +
$$
$$
2C_{\text{lip}}^{2/3}C_1^{2/3}C_\rho^{1/6} \frac{d^{1+1/4}L^{1+q}\log^{2/3}L}{N^{1/(4+2\delta)}}
$$
$$
+ 2C_{\text{lip}}^{1/3}C_1^{1/3}C_\rho^{1/3} \frac{dL^{1/2+q}\log^{2/3}L}{\sqrt{N}}
$$
$$
+ 2C_{\text{lip}}^{1/2}C_1^{1/2}C_2^{1/4} \frac{d^{1+1/8}L^{3/4+q}\log L}{N^{1/(4+2\delta)}} \Bigg). \tag{B.72}
$$

From (B.72), we see that if we choose $n$ such that
$$
n \geq (2C_{\text{diff}})^{12} L^{60+44q}d^3
$$
and we choose $N$ such that
$$
N^{1/(2+\delta)} \geq 16C_{\text{Lip}}^{4/3}C_1^{4/3}C_2^{1/2}C_\rho^{1/3}d^{1/2}L^{2+2q}
$$
then (B.72) implies the bound
$$
\Lambda_k \leq 3CC_1 d^{5/4}L^{3/2}\log^2 L,
$$
which agrees with the previous choice $C_{\text{lip}} = 3C$ and thus proves (B.65) in the remaining case of (B.71). By induction, then, we have proved that (B.64) and (B.65) hold for each index $0 \leq k \leq \lfloor L^q/(n\tau) \rfloor$.

We can now wrap up the proof: we will obtain the desired conclusion by plugging the results we have developed into (B.57) and simplifying. Plugging (B.37), (B.27) and (B.64) into (B.49) and bounding the maximum by the sum, we get

$$
\left\| \zeta_{\lfloor L^q/(n\tau) \rfloor}^{N,\text{Lip}} \right\|_{L^2_{\mu^\infty}(\mathcal{M})}
$$
$$
\leq \left\| \zeta_{\lfloor L^q/(n\tau) \rfloor}^{\infty} \right\|_{L^2_{\mu^\infty}(\mathcal{M})} + \left\| \zeta_{\lfloor L^q/(n\tau) \rfloor}^{N} - \zeta_{\lfloor L^q/(n\tau) \rfloor}^{\infty} \right\|_{L^2_{\mu^\infty}(\mathcal{M})} + \left\| \delta_{\lfloor L^q/(n\tau) \rfloor}^{N} \right\|_{L^\infty(\mathcal{M})}
$$
$$
\leq \begin{array}{l} \frac{CC_\rho^{q_{\text{cert}}}\sqrt{d}\log L}{n\tau\lfloor L^q/(n\tau) \rfloor} + C' \left( \frac{L^{60+32q}d^{15}\log^9 L}{n} \right)^{1/12} \\ \quad + C''C_1C_2^{1/2}C_\rho^{4/3}\frac{d^{7/4}L^{5/2+q}\log^2 L}{N^{1/(2+\delta)}} \end{array}
$$
$$
\leq \frac{CC_\rho^{q_{\text{cert}}}\sqrt{d}\log L}{L^q} + C' \left( \frac{L^{60+32q}d^{15}\log^9 L}{n} \right)^{1/12} + C''C_1C_2^{1/2}C_\rho^{4/3}\frac{d^{7/4}L^{5/2+q}\log^2 L}{N^{1/(2+\delta)}}
$$
$$
\leq \frac{CC_\rho^{q_{\text{cert}}}\sqrt{d}\log L}{L^q}, \tag{B.73}
$$

where in the third inequality we apply $\lfloor L^q/(n\tau) \rfloor \geq L^q/(2n\tau)$, which follows from our choice of step size, and in the fourth inequality we simplify residuals using $n \geq (C'/C)^{12}d^9L^{60+44q}$ and $N^{1/(2+\delta)} \geq C''C_1C_2^{1/2}C_\rho^{4/3}d^{5/4}L^{5/2+2q}\log L$. Applying (B.73), the triangle inequality (with (B.37) and the fact that $\mu^\infty$ is a probability measure) and our previous choice of large $n$, we get

$$
\left\| \zeta_{\lfloor L^q/(n\tau) \rfloor}^{N} \right\|_{L^2_{\mu^\infty}(\mathcal{M})} \leq \frac{CC_\rho^{q_{\text{cert}}}\sqrt{d}\log L}{L^q}, \tag{B.74}
$$

i.e. generalization in $L^2_{\mu^\infty}(\mathcal{M})$. We can bootstrap generalization in $L^\infty(\mathcal{M})$ from (B.73) using the triangle inequality and (B.57): we get

$$
\left\| \zeta_{\lfloor L^q/(n\tau) \rfloor}^{N} \right\|_{L^\infty(\mathcal{M})}
$$
$$
\leq CC_2^{1/2}\left\| \zeta_{\lfloor L^q/(n\tau) \rfloor}^{N,\text{Lip}} \right\|_{L^2_{\mu^\infty}(\mathcal{M})} + \frac{C}{\rho_{\min}^{1/3}}\left\| \zeta_{\lfloor L^q/(n\tau) \rfloor}^{N,\text{Lip}} \right\|_{L^2_{\mu^\infty}(\mathcal{M})}^{2/3} \Lambda_{\lfloor L^q/(n\tau) \rfloor}^{1/3} + \left\| \delta_{\lfloor L^q/(n\tau) \rfloor}^{N} \right\|_{L^\infty(\mathcal{M})}
$$
$$
\leq \frac{CC_2^{1/2}C_\rho^{q_{\text{cert}}}\sqrt{d}\log L}{L^q} + \frac{C'C_1^{1/3}}{\rho_{\min}^{1/3}}\frac{d^{3/4}L^{(3-4q)/6}\log^{4/3}L}{\min\{\rho_{\min}^{1/3},1\}},
$$

where in the second line we apply (B.37) and our previous choice of large $n$ to absorb the residual from $\delta^N_{\lfloor L^q/(n\tau)\rfloor}$, and apply (B.65) to bound the $\Lambda^{1/3}_{\lfloor L^q/(n\tau)\rfloor}$ term. Worst-casing the errors in the previous bound, we obtain

$$\left\| \zeta^N_{\lfloor L^q/(n\tau)\rfloor} \right\|_{L^\infty(\mathcal{M})} \le C \left( C_\rho^{q_{\text{cert}}} C_2^{1/2} + C_1^{1/3} C_\rho^{2/3} \right) \frac{d^{3/4} \log^{4/3} L}{L^{(4q-3)/6}}.$$

To conclude, we will tally dependencies and make some simplifications to show the conditions stated in the result suffice. Recalling (B.53) and (B.58) and using (B.62), we have

$$C_1 \le CC_\rho^{2q_{\text{cert}}+1} (1 + \rho_{\max})^{1/2} e^{14/\delta},$$

so we can simplify to

$$C_\rho^{1/2} C_2^{1/2} + C_1^{1/3} C_\rho^{2/3} \le C_\rho \left( \frac{\rho_{\max}}{\min\{\mu^\infty(\mathcal{M}_+), \mu^\infty(\mathcal{M}_-)\}} \right)^{1/2}$$
$$+ CC_\rho^{1+2q_{\text{cert}}/3} (1 + \rho_{\max})^{1/6} e^{14/(3\delta)}$$
$$\le \frac{CC_\rho^{1+2q_{\text{cert}}/3} (1 + \rho_{\max})^{1/2} e^{14/(3\delta)}}{\min\{\mu^\infty(\mathcal{M}_+), \mu^\infty(\mathcal{M}_-)\}^{1/2}}.$$

We can use this to obtain a simplified generalization in $L^\infty(\mathcal{M})$ bound from our previous expression: it becomes

$$\left\| \zeta^N_{\lfloor L^q/(n\tau)\rfloor} \right\|_{L^\infty(\mathcal{M})} \le \frac{CC_\rho^{1+2q_{\text{cert}}/3} (1 + \rho_{\max})^{1/2} e^{14/(3\delta)}}{\min\{\mu^\infty(\mathcal{M}_+), \mu^\infty(\mathcal{M}_-)\}^{1/2}} \frac{d^{3/4} \log^{4/3} L}{L^{(4q-3)/6}}, \tag{B.75}$$

which can be made nonvacuous when $q > 3/4$. Tallying dependencies, we find after worst-casing (and using $q \ge 1/2$ and some interdependences between parameters to simplify) that it suffices to choose $N$ such that

$$N^{1/(2+\delta)} \ge CC_1^{4/3} C_2^{1/2} C_\rho^{5/3} e^{21/\delta} d^{5/4} L^{5/2+2q} \log L,$$

the depth $L$ such that

$$L \ge C \max\{C_\rho^{2q_{\text{cert}}} d, \kappa^2 C_\lambda\},$$

the width $n$ such that

$$n \ge C \max\left\{ e^{252/\delta} L^{60+44q} d^9 \log^9 L, \kappa^{2/5}, \left( \frac{\kappa}{c_\lambda} \right)^{1/3} \right\},$$

and $d$ such that $d \ge Cd_0 \log(nn_0 C_{\mathcal{M}})$. Unpacking the constants in the condition on $N$, we see that it suffices to choose $N$ such that

$$N^{1/(2+\delta)} \ge \frac{CC_\rho^{7/2+8q_{\text{cert}}/3} (1 + \rho_{\max})^{7/6} e^{119/(3\delta)}}{\min\{\mu^\infty(\mathcal{M}_+), \mu^\infty(\mathcal{M}_-)\}^{1/2}} d^{5/4} L^{5/2+2q} \log L.$$

$\square$

## B.3 AUXILIARY RESULTS

**Lemma B.8.** *Defining a kernel*

$$\Theta^N_k(\boldsymbol{x}, \boldsymbol{x}') = \int_0^1 \left\langle \widetilde{\nabla} f_{\boldsymbol{\theta}^N_k}(\boldsymbol{x}'), \widetilde{\nabla} f_{\boldsymbol{\theta}^N_k - t\tau \widetilde{\nabla} \mathcal{L}_{\mu^N}(\boldsymbol{\theta}^N_k)}(\boldsymbol{x}) \right\rangle \mathrm{d}t$$

*and corresponding operator on $L^2_{\mu^N}(\mathcal{M})$*

$$\boldsymbol{\Theta}^N_k[g](\boldsymbol{x}) = \int_{\mathcal{M}} \Theta^N_k(\boldsymbol{x}, \boldsymbol{x}') g(\boldsymbol{x}') \, \mathrm{d}\mu^N(\boldsymbol{x}'),$$

*we have that $\boldsymbol{\Theta}^N_k$ is bounded, and*

$$\zeta^N_{k+1} = \left( \mathrm{Id} - \tau \boldsymbol{\Theta}^N_k \right) \left[ \zeta^N_k \right].$$

*Proof.* By the definition of the gradient iteration, we have that

$$\zeta_{k+1}^N - \zeta_k^N = f_{\boldsymbol{\theta}_k^N - \tau \widetilde{\nabla} \mathcal{L}_{\mu^N}(\boldsymbol{\theta}_k^N)} - f_{\boldsymbol{\theta}_k^N}.$$

The total number of trainable parameters in the network is $M = n(n(L-1) + n_0 + 1)$, and the euclidean space in which $\boldsymbol{\theta}$ lies is isomorphic to $\mathbb{R}^M$. For $k \in \mathbb{N}_0$, define paths $\boldsymbol{\gamma}_k^N : [0, 1] \to \mathbb{R}^M$ by

$$\boldsymbol{\gamma}_k^N(t) = \boldsymbol{\theta}_k^N - t\tau \widetilde{\nabla} \mathcal{L}_{\mu^N}(\boldsymbol{\theta}_k^N),$$

so that

$$\zeta_{k+1}^N - \zeta_k^N = f_{\boldsymbol{\gamma}_k^N(1)} - f_{\boldsymbol{\gamma}_k^N(0)}.$$

We will justify a first-order Taylor representation in integral form based on the previous expression by arguing that for every $\boldsymbol{x} \in \mathcal{M}$, $t \mapsto f_{\boldsymbol{\gamma}_k^N(t)}(\boldsymbol{x})$ is absolutely continuous on $[0, 1]$, by checking the hypotheses of (Cohn, 2013, Theorem 6.3.11). Because $\boldsymbol{\gamma}_k^N$ is smooth and $f_{(\cdot)}(\boldsymbol{x})$ is continuous, $f_{\boldsymbol{\gamma}_k^N(t)}$ is also continuous. Continuity of the features as a function of the parameters and of $\boldsymbol{\gamma}_k^N$ implies that for every $\ell \geq 0$, the image of $[0, 1]$ under the map

$$t \mapsto \boldsymbol{\alpha}_{\boldsymbol{\gamma}_k^N(t)}^\ell(\boldsymbol{x})$$

is compact. By repeated application of Lemma E.21, we conclude that $t \mapsto f_{\boldsymbol{\gamma}_k^N(t)}(\boldsymbol{x})$ is differentiable at all but countably many points of $[0, 1]$. Following the proof of Lemma E.21, we see that the points of nondifferentiability of $t \mapsto f_{\boldsymbol{\gamma}_k^N(t)}(\boldsymbol{x})$ are contained in the set of points of $[0, 1]$ where there exists a layer $\ell$ at which at least one of the coordinates of $\boldsymbol{\alpha}_{\boldsymbol{\gamma}_k^N(\cdot)}^\ell(\boldsymbol{x})$ vanishes. Applying the chain rule at points of differentiability of the ReLU $[\cdot]_+$ and assigning $0$ otherwise, it follows that the derivative of $t \mapsto f_{\boldsymbol{\gamma}_k^N(t)}(\boldsymbol{x})$ at $t \in [0, 1]$ is equal to

$$-\tau \left\langle \widetilde{\nabla} \mathcal{L}_{\mu^N}(\boldsymbol{\theta}_k^N), \widetilde{\nabla} f_{\boldsymbol{\gamma}_k^N(t)}(\boldsymbol{x}) \right\rangle$$

at all but countably many points $t \in [0, 1]$. We finally need to check integrability of this derivative on $[0, 1]$. We have by linearity

$$-\tau \left\langle \widetilde{\nabla} \mathcal{L}_{\mu^N}(\boldsymbol{\theta}_k^N), \widetilde{\nabla} f_{\boldsymbol{\gamma}_k^N(t)}(\boldsymbol{x}) \right\rangle = -\tau \int_{\mathcal{M}} \zeta_{\boldsymbol{\theta}_k^N}(\boldsymbol{x}') \left\langle \widetilde{\nabla} f_{\boldsymbol{\theta}_k^N}(\boldsymbol{x}'), \widetilde{\nabla} f_{\boldsymbol{\gamma}_k^N(t)}(\boldsymbol{x}) \right\rangle \mathrm{d}\mu^N(\boldsymbol{x}'), \quad \text{(B.76)}$$

and by definition

$$\left\langle \widetilde{\nabla} f_{\boldsymbol{\gamma}_k^N(t)}(\boldsymbol{x}), \widetilde{\nabla} f_{\boldsymbol{\theta}_k^N}(\boldsymbol{x}') \right\rangle$$

$$= \left\langle \boldsymbol{\alpha}_{\boldsymbol{\gamma}_k^N(t)}^L(\boldsymbol{x}), \boldsymbol{\alpha}_{\boldsymbol{\theta}_k^N}^L(\boldsymbol{x}') \right\rangle + \sum_{\ell=0}^{L-1} \left\langle \boldsymbol{\alpha}_{\boldsymbol{\gamma}_k^N(t)}^\ell(\boldsymbol{x}), \boldsymbol{\alpha}_{\boldsymbol{\theta}_k^N}^\ell(\boldsymbol{x}') \right\rangle \left\langle \boldsymbol{\beta}_{\boldsymbol{\gamma}_k^N(t)}^\ell(\boldsymbol{x}), \boldsymbol{\beta}_{\boldsymbol{\theta}_k^N}^\ell(\boldsymbol{x}') \right\rangle.$$

By construction of the network, the feature maps $(t, \boldsymbol{x}) \mapsto \boldsymbol{\alpha}_{\boldsymbol{\gamma}_k^N(t)}^\ell(\boldsymbol{x})$ are continuous. For the backward feature maps, we can write for any $\boldsymbol{\theta}_1 = (\boldsymbol{W}_1^1, \ldots, \boldsymbol{W}_1^{L+1})$ and any $\boldsymbol{\theta}_2 = (\boldsymbol{W}_2^1, \ldots, \boldsymbol{W}_2^{L+1})$ using Cauchy-Schwarz

$$\left| \langle \boldsymbol{\beta}_{\boldsymbol{\theta}_1}^\ell(\boldsymbol{x}), \boldsymbol{\beta}_{\boldsymbol{\theta}_2}^\ell(\boldsymbol{x}') \rangle \right| \leq \prod_{\ell'=\ell+1}^{L} \left\| \boldsymbol{W}_1^{\ell'+1} \right\| \left\| \boldsymbol{W}_2^{\ell'+1} \right\|,$$

and the RHS of this bound is a continuous function of $(\boldsymbol{\theta}, \boldsymbol{x})$. Because our domain of interest $[0, 1] \times \mathcal{M}$ is compact, we have from the triangle inequality, the previous bound on the backward feature inner products and the Weierstrass theorem

$$\sup_{t \in [0,1], \, \boldsymbol{x} \in \mathcal{M}} \left| \left\langle \widetilde{\nabla} f_{\boldsymbol{\gamma}_k^N(t)}(\boldsymbol{x}), \widetilde{\nabla} f_{\boldsymbol{\theta}_k^N}(\boldsymbol{x}') \right\rangle \right| < +\infty, \quad \text{(B.77)}$$

so that in particular, we can bound our expression for the derivative of $t \mapsto f_{\boldsymbol{\gamma}_k^N(t)}(\boldsymbol{x})$ using the triangle inequality as

$$\left| -\tau \left\langle \widetilde{\nabla} \mathcal{L}_{\mu^N}(\boldsymbol{\theta}_k^N), \widetilde{\nabla} f_{\boldsymbol{\gamma}_k^N(t)}(\boldsymbol{x}) \right\rangle \right| \leq C\tau \int_{\mathcal{M}} \left| \zeta_{\boldsymbol{\theta}_k^N}(\boldsymbol{x}') \right| \mathrm{d}\mu^N(\boldsymbol{x}')$$

for some constant $C > 0$. The RHS of the previous bound does not depend on $t$, so by an application of (Cohn, 2013, Theorem 6.3.11), it follows that $t \mapsto f_{\gamma_k^N(t)}(\boldsymbol{x})$ is absolutely continuous, and we have the representation

$$\zeta_{k+1}^N(\boldsymbol{x}) - \zeta_k^N(\boldsymbol{x}) = -\tau \int_0^1 \left\langle \widetilde{\nabla} \mathcal{L}_{\mu^N}(\boldsymbol{\theta}_k^N), \widetilde{\nabla} f_{\gamma_k^N(t)}(\boldsymbol{x}) \right\rangle \mathrm{d}t.$$

Using (B.76), we can express this as

$$\zeta_{k+1}^N(\boldsymbol{x}) - \zeta_k^N(\boldsymbol{x}) = -\tau \int_0^1 \left( \int_{\mathcal{M}} \zeta_{\boldsymbol{\theta}_k^N}(\boldsymbol{x}') \left\langle \widetilde{\nabla} f_{\boldsymbol{\theta}_k^N}(\boldsymbol{x}'), \widetilde{\nabla} f_{\gamma_k^N(t)}(\boldsymbol{x}) \right\rangle \mathrm{d}\mu^N(\boldsymbol{x}') \right) \mathrm{d}t.$$

To conclude, it will be convenient to switch the order of integration appearing in the previous expression. Applying (B.77), we have

$$\left| \zeta_{\boldsymbol{\theta}_k^N}(\boldsymbol{x}') \left\langle \widetilde{\nabla} f_{\boldsymbol{\theta}_k^N}(\boldsymbol{x}'), \widetilde{\nabla} f_{\gamma_k^N(t)}(\boldsymbol{x}) \right\rangle \right| \leq C \left| \zeta_{\boldsymbol{\theta}_k^N}(\boldsymbol{x}') \right|,$$

and the RHS of this bound is integrable over $[0,1] \times \mathcal{M}$ because the network is a continuous function of the input. By Fubini's theorem, it follows

$$\zeta_{k+1}^N(\boldsymbol{x}) - \zeta_k^N(\boldsymbol{x}) = -\tau \int_{\mathcal{M}} \left( \int_0^1 \left\langle \widetilde{\nabla} f_{\boldsymbol{\theta}_k^N}(\boldsymbol{x}'), \widetilde{\nabla} f_{\gamma_k^N(t)}(\boldsymbol{x}) \right\rangle \mathrm{d}t \right) \zeta_{\boldsymbol{\theta}_k^N}(\boldsymbol{x}') \, \mathrm{d}\mu^N(\boldsymbol{x}') \qquad \text{(B.78)}$$

Defining

$$\Theta_k^N(\boldsymbol{x}, \boldsymbol{x}') = \int_0^1 \left\langle \widetilde{\nabla} f_{\boldsymbol{\theta}_k^N}(\boldsymbol{x}'), \widetilde{\nabla} f_{\gamma_k^N(t)}(\boldsymbol{x}) \right\rangle \mathrm{d}t$$

and using (B.77), we can define bounded operators $\Theta_k^N : L^2_{\mu^N}(\mathcal{M}) \to L^2_{\mu^N}(\mathcal{M})$ by

$$\boldsymbol{\Theta}_k^N[g](\boldsymbol{x}) = \int_{\mathcal{M}} \Theta_k^N(\boldsymbol{x}, \boldsymbol{x}') g(\boldsymbol{x}') \, \mathrm{d}\mu^N(\boldsymbol{x}'),$$

and with this definition, (B.78) becomes

$$\zeta_{k+1}^N - \zeta_k^N = -\tau \boldsymbol{\Theta}_k^N \left[ \zeta_k^N \right],$$

as claimed. $\qquad \square$

**Lemma B.9.** *For any network parameters $\boldsymbol{\theta}$, define kernels*

$$\Theta_{\boldsymbol{\theta}}(\boldsymbol{x}, \boldsymbol{x}') = \left\langle \widetilde{\nabla} f_{\boldsymbol{\theta}}(\boldsymbol{x}'), \widetilde{\nabla} f_{\boldsymbol{\theta}}(\boldsymbol{x}) \right\rangle,$$

*and for $\star \in \{N, \infty\}$, define corresponding bounded operators on $L^2_{\mu^\star}(\mathcal{M})$ by*

$$\boldsymbol{\Theta}_{\boldsymbol{\theta}, \mu^\star}[g](\boldsymbol{x}) = \int_{\mathcal{M}} \Theta_{\boldsymbol{\theta}}(\boldsymbol{x}, \boldsymbol{x}') g(\boldsymbol{x}') \, \mathrm{d}\mu^\star(\boldsymbol{x}').$$

*For any settings of the parameters $\boldsymbol{\theta}$, the operators $\boldsymbol{\Theta}_{\boldsymbol{\theta}, \mu^\star}$ are self-adjoint, positive, and compact. In particular, they diagonalize in a countable orthonormal basis of $L^2_{\mu^\star}(\mathcal{M})$ functions with corresponding nonnegative eigenvalues.*

*Proof.* When $\star = N$, an identification reduces the operators $\boldsymbol{\Theta}_{\boldsymbol{\theta}, \mu^\star}$ to operators on finite-dimensional vector spaces, and the claims follow immediately from general principles and the finite-dimensional spectral theorem. We therefore only work out the details for the case $\star = \infty$. Boundedness follows from an argument identical to the one developed in the proof of Lemma B.8, in particular to develop an estimate analogous to (B.77). This estimate, together with separability and compactness of $\mathcal{M}$, also establishes that $\boldsymbol{\Theta}_{\boldsymbol{\theta}, \infty}$ is compact, by standard results for Hilbert-Schmidt operators (Heil, 2011, §B). In addition, this estimate allows us to apply Fubini's theorem to write for any $g_1, g_2 \in L^2_{\mu^\infty}(\mathcal{M})$

$$\langle g_1, \boldsymbol{\Theta}_{\boldsymbol{\theta}, \infty}[g_2] \rangle_{L^2_{\mu^\infty}(\mathcal{M})} = \iint_{\mathcal{M} \times \mathcal{M}} \Theta_{\boldsymbol{\theta}}(\boldsymbol{x}, \boldsymbol{x}') g_1(\boldsymbol{x}) g_2(\boldsymbol{x}') \, \mathrm{d}\mu^\infty(\boldsymbol{x}) \, \mathrm{d}\mu^\infty(\boldsymbol{x}') = \langle g_2, \boldsymbol{\Theta}_{\boldsymbol{\theta}, \infty}[g_1] \rangle$$

since $\Theta_{\boldsymbol{\theta}}(\boldsymbol{x}, \boldsymbol{x}') = \Theta_{\boldsymbol{\theta}}(\boldsymbol{x}', \boldsymbol{x})$. A similar calculation establishes positivity: we have for any $g \in L^2_{\mu^\infty}(\mathcal{M})$

$$
\begin{aligned}
\langle g, \boldsymbol{\Theta}_{\boldsymbol{\theta}, \infty}[g] \rangle_{L^2_{\mu^\infty}(\mathcal{M})} &= \iint_{\mathcal{M} \times \mathcal{M}} \left\langle \widetilde{\nabla} f_{\boldsymbol{\theta}}(\boldsymbol{x}'), \widetilde{\nabla} f_{\boldsymbol{\theta}}(\boldsymbol{x}) \right\rangle g(\boldsymbol{x}) g(\boldsymbol{x}') \, \mathrm{d}\mu^\infty(\boldsymbol{x}) \, \mathrm{d}\mu^\infty(\boldsymbol{x}') \\
&= \left\langle \int_{\mathcal{M}} \widetilde{\nabla} f_{\boldsymbol{\theta}}(\boldsymbol{x}) g(\boldsymbol{x}) \, \mathrm{d}\mu^\infty(\boldsymbol{x}), \int_{\mathcal{M}} \widetilde{\nabla} f_{\boldsymbol{\theta}}(\boldsymbol{x}) g(\boldsymbol{x}) \, \mathrm{d}\mu^\infty(\boldsymbol{x}) \right\rangle \geq 0,
\end{aligned}
$$

where we applied Fubini's theorem and linearity of the integral. These facts and the spectral theorem for self-adjoint compact operators on a Hilbert space imply in particular that the operator $\boldsymbol{\Theta}_{\boldsymbol{\theta}, \infty}$ can be diagonalized in a countable orthonormal basis of eigenfunctions $(v_i)_{i \in \mathbb{N}} \subset L^2_{\mu^\infty}(\mathcal{M})$ with corresponding nonnegative eigenvalues $(\lambda_i)_{i \in \mathbb{N}} \subset [0, +\infty)$. $\qquad\square$

**Lemma B.10.** *Write $\boldsymbol{\Theta}_{\mu^N}$ for the operator defined in Lemma B.9, with the parameters $\boldsymbol{\theta}$ set to the initial random network weights and the measure set to $\mu^N$. There exist absolute constants $c, K, K' > 0$ such that for any $q \geq 0$ and any $d \geq K d_0 \log(n n_0 C_{\mathcal{M}})$, if*

$$
\tau < \frac{1}{\left\| \boldsymbol{\Theta}_{\mu^N} \right\|_{L^2_{\mu^N}(\mathcal{M}) \to L^2_{\mu^N}(\mathcal{M})}}
$$

*and if in addition $n \geq K' L^{48 + 20q} d^9 \log^9 L$, then one has*

$$
\mathbb{P}\left[ \bigcap_{0 \leq k \leq L^q/(n\tau)} \left\{ \left\| \zeta_i^N \right\|_{L^2_{\mu^N}(\mathcal{M})} \leq \sqrt{d} \right\} \right] \geq 1 - \left( 1 + \frac{2L^q}{n\tau} \right) e^{-cd}.
$$

*Proof.* Consider the nominal error evolution $\zeta_k^{N,\mathrm{nom}}$, defined iteratively as

$$
\begin{aligned}
\zeta_{k+1}^{N,\mathrm{nom}} &= \zeta_k^{N,\mathrm{nom}} - \tau \boldsymbol{\Theta}_{\mu^N} \left[ \zeta_k^{N,\mathrm{nom}} \right]; \\
\zeta_0^{N,\mathrm{nom}} &= \zeta
\end{aligned}
$$

for a step size $\tau > 0$, which satisfies

$$
\tau < \frac{1}{\left\| \boldsymbol{\Theta}_{\mu^N} \right\|_{L^2_{\mu^N}(\mathcal{M}) \to L^2_{\mu^N}(\mathcal{M})}}.
$$

We will prove the claim by showing that this auxiliary iteration is monotone decreasing in the loss, and close enough to the gradient-like iteration of interest that we can prove that the gradient-like iteration also retains a controlled loss. These dynamics satisfy the 'update equation'

$$
\zeta_k^{N,\mathrm{nom}} = \left( \mathrm{Id} - \tau \boldsymbol{\Theta}_{\mu^N} \right)^k [\zeta].
$$

Because $\mathcal{M}$ is compact and $\zeta$ is a continuous function of the input, we have $\zeta \in L^\infty(\mathcal{M})$ for all values of the random weights. Because $\mu^N$ is a probability measure, this means $\zeta$ has finite $L^p_{\mu^N}(\mathcal{M})$ norm for every $p > 0$. Meanwhile, the choice of $\tau$ and positivity of the operator (by Lemma B.9) guarantees

$$
\left\| \mathrm{Id} - \tau \boldsymbol{\Theta}_{\mu^N} \right\|_{L^2_{\mu^N}(\mathcal{M}) \to L^2_{\mu^N}(\mathcal{M})} \leq 1,
$$

from which it follows from the update equation

$$
\left\| \zeta_k^{N,\mathrm{nom}} \right\|_{L^2_{\mu^N}(\mathcal{M})} \leq \|\zeta\|_{L^2_{\mu^N}(\mathcal{M})} \leq \|\zeta\|_{L^\infty(\mathcal{M})}, \tag{B.79}
$$

where the last inequality uses that $\mu^N$ is a probability measure. In particular, this nominal error evolution is nonincreasing in the relevant loss. Now, we recall the update equation for the finite-sample dynamics

$$
\zeta_{k+1}^N = \left( \mathrm{Id} - \tau \boldsymbol{\Theta}^N \right) \left[ \zeta_k^N \right],
$$

which follows from Lemma B.8. Subtracting and rearranging, this gives an update equation for the difference:

$$
\zeta_{k+1}^N - \zeta_{k+1}^{N,\mathrm{nom}} = \left( \mathrm{Id} - \tau \boldsymbol{\Theta}_{\mu^N} \right) \left[ \zeta_k^N - \zeta_k^{N,\mathrm{nom}} \right] - \tau \left( \boldsymbol{\Theta}_k^N - \boldsymbol{\Theta}_{\mu^N} \right) \left[ \zeta_k^N \right]. \tag{B.80}
$$

Under our hypothesis on $\tau$, (B.80) and the triangle inequality imply the bound

$$
\left\|\zeta_{k+1}^N - \zeta_{k+1}^{N,\mathrm{nom}}\right\|_{L^2_{\mu^N}(\mathcal{M})}
$$
$$
\leq \left\|\zeta_k^N - \zeta_k^{N,\mathrm{nom}}\right\|_{L^2_{\mu^N}(\mathcal{M})} + \tau\left\|\zeta_k^N\right\|_{L^2_{\mu^N}(\mathcal{M})}\left\|\boldsymbol{\Theta}_k^N - \boldsymbol{\Theta}_{\mu^N}\right\|_{L^2_{\mu^N}(\mathcal{M})\to L^2_{\mu^N}(\mathcal{M})}.
$$

Using Jensen's inequality and the Schwarz inequality, we have

$$
\left\|\boldsymbol{\Theta}_k^N - \boldsymbol{\Theta}_{\mu^N}\right\|_{L^2_{\mu^N}(\mathcal{M})\to L^2_{\mu^N}(\mathcal{M})}
$$
$$
\leq \sup_{\|g\|_{L^2_{\mu^N}(\mathcal{M})}\leq 1} \int_{\mathcal{M}}\left\|\Theta_k^N(\,\cdot\,,\boldsymbol{x}') - \Theta(\,\cdot\,,\boldsymbol{x}')\right\|_{L^2_{\mu^N}(\mathcal{M})}|g(\boldsymbol{x}')|\,\mathrm{d}\mu^N(\boldsymbol{x}')
$$
$$
\leq \sup_{\|g\|_{L^2_{\mu^N}(\mathcal{M})}\leq 1}\left\|\Theta_k^N - \Theta\right\|_{L^\infty(\mathcal{M}\times\mathcal{M})}\|g\|_{L^1_{\mu^N}(\mathcal{M})}
$$
$$
\leq \left\|\Theta_k^N - \Theta\right\|_{L^\infty(\mathcal{M}\times\mathcal{M})},
$$

since $\mu^N$ is a probability measure. Defining

$$
\Delta_k^N = \max_{i\in\{0,1,\dots,k\}}\left\|\Theta_i^N - \Theta\right\|_{L^\infty(\mathcal{M}\times\mathcal{M})},
$$

by a telescoping series and the identical initial conditions, we thus obtain

$$
\left\|\zeta_{k+1}^N - \zeta_{k+1}^{N,\mathrm{nom}}\right\|_{L^2_{\mu^N}(\mathcal{M})} \leq \tau\Delta_k^N\sum_{i=0}^{k}\left\|\zeta_i^N\right\|_{L^2_{\mu^N}(\mathcal{M})},
$$

and the triangle inequality and (B.79) then yield

$$
\left\|\zeta_{k+1}^N\right\|_{L^2_{\mu^N}(\mathcal{M})} \leq \|\zeta\|_{L^\infty(\mathcal{M})} + \tau\Delta_k^N\sum_{i=0}^{k}\left\|\zeta_i^N\right\|_{L^2_{\mu^N}(\mathcal{M})}.
$$

Using a discrete version of (the standard) Gronwall's inequality, the previous bound implies

$$
\left\|\zeta_k^N\right\|_{L^2_{\mu^N}(\mathcal{M})} \leq \|\zeta\|_{L^\infty(\mathcal{M})} + \|\zeta\|_{L^\infty(\mathcal{M})}\sum_{i=0}^{k-1}\tau\Delta_{k-1}^N\exp\left(\sum_{j=i+1}^{k-1}\tau\Delta_{k-1}^N\right)
$$
$$
\leq \|\zeta\|_{L^\infty(\mathcal{M})}\left(1 + k\tau\Delta_{k-1}^N\exp\left(k\tau\Delta_{k-1}^N\right)\right). \tag{B.81}
$$

To conclude, we will use Lemma F.5 and an inductive argument based on (B.81). Let us first observe that by Lemma D.11 (with a rescaling of $d$, which worsens the absolute constants), we have

$$
\mathbb{P}\left[\|\zeta\|_{L^\infty(\mathcal{M})} \leq \frac{\sqrt{d}}{2}\right] \geq 1 - e^{-cd} \tag{B.82}
$$

as long as $n \geq Kd^4L$ and $d \geq K'd_0\log(nn_0C_{\mathcal{M}})$. Define events $\mathcal{E}_k^N$ by

$$
\mathcal{E}_k^N = \left\{\left\|\zeta_k^N\right\|_{L^2_{\mu^N}(\mathcal{M})} > \sqrt{d}\right\},
$$

where $d > 0$ is sufficiently large to satisfy the conditions on $d$ given above. We are interested in controlling the probability of $\bigcup_{i=0}^{k}\mathcal{E}_k^N$ for $k\in\mathbb{N}_0$. We can write

$$
\mathbb{P}\left[\bigcup_{i=0}^{k}\mathcal{E}_i^N\right] = \mathbb{P}\left[\bigcup_{i=0}^{k-1}\mathcal{E}_i^N\right] + \mathbb{P}\left[\mathcal{E}_k^N\cap\bigcap_{i=0}^{k-1}\left(\mathcal{E}_i^N\right)^{\mathsf{c}}\right],
$$

and unraveling, we obtain

$$
\mathbb{P}\left[\bigcup_{i=0}^{k}\mathcal{E}_k^N\right] = \sum_{i=0}^{k}\mathbb{P}\left[\mathcal{E}_i^N\cap\bigcap_{j=0}^{i-1}\left(\mathcal{E}_j^N\right)^{\mathsf{c}}\right].
$$

In words, it is enough to control the sum of the measures of the parts of $\mathcal{E}_k^N$ that are common with the part of the space where none of the past events occurs. First, note that (B.82) implies

$$\mathbb{P}\big[\mathcal{E}_0^N\big] \le e^{-cd},$$

and so assume $i > 0$ below. For any $q > 0$, if $k\tau \le L^q/n$, $n \ge KL^{36+8q}d^9$ and $d \ge K'd_0 \log(nn_0 C_{\mathcal{M}})$, Lemma F.5 gives that there are events $\mathcal{B}_i^N$ that respectively contain the sets $\{\Delta_{i-1}^N > CL^{4+2q/3}d^{3/4}n^{11/12}\log^{3/4}L\}$, and which satisfy in addition

$$\mathbb{P}\left[\mathcal{B}_i^N \cap \bigcap_{j=0}^{i-1}\big(\mathcal{E}_j^N\big)^{\mathsf{c}}\right] \le e^{-cd}.$$

We thus have by this last bound, a partition, and intersection monotonicity

$$\mathbb{P}\left[\mathcal{E}_i^N \cap \bigcap_{j=0}^{i-1}\big(\mathcal{E}_j^N\big)^{\mathsf{c}}\right] \le e^{-cd} + \mathbb{P}\left[\mathcal{E}_i^N \cap \big(\mathcal{B}_i^N\big)^{\mathsf{c}}\right],$$

and by construction, one has $\Delta_{i-1}^N \le CL^{4+2q/3}d^{3/4}n^{11/12}\log^{3/4}L$ on $\big(\mathcal{B}_i^N\big)^{\mathsf{c}}$. Another partition and (B.82) give

$$\mathbb{P}\left[\mathcal{E}_i^N \cap \big(\mathcal{B}_i^N\big)^{\mathsf{c}}\right] \le e^{-cd} + \mathbb{P}\left[\mathcal{E}_i^N \cap \big(\mathcal{B}_i^N\big)^{\mathsf{c}} \cap \left\{\|\zeta\|_{L^\infty(\mathcal{M})} \le \frac{\sqrt{d}}{2}\right\}\right].$$

When the two events on the RHS of the last bound are active, we can obtain using (B.81)

$$\big\|\zeta_k^N\big\|_{L_{\mu^N}^2(\mathcal{M})}$$
$$\le \frac{\sqrt{d}}{2}\left(1 + k\tau CL^{4+2q/3}d^{3/4}n^{11/12}\log^{3/4}L \exp\left(k\tau CL^{4+2q/3}d^{3/4}n^{11/12}\log^{3/4}L\right)\right).$$

Given that $k\tau \le L^q/n$, we have

$$k\tau CL^{4+2q/3}d^{3/4}n^{11/12}\log^{3/4}L \le \left(\frac{C^{12}L^{48+20q}d^9\log^9 L}{n}\right)^{1/12} \le 1/e,$$

where the last bound holds provided $n \ge KL^{48+20q}d^9\log^9 L$ . Thus, on the event

$$\big(\mathcal{B}_i^N\big)^{\mathsf{c}} \cap \left\{\|\zeta\|_{L^\infty(\mathcal{M})} \le \frac{\sqrt{d}}{2}\right\},$$

we have

$$\big\|\zeta_k^N\big\|_{L_{\mu^N}^2(\mathcal{M})} \le \sqrt{d},$$

and thus

$$\mathbb{P}\left[\mathcal{E}_i^N \cap \big(\mathcal{B}_i^N\big)^{\mathsf{c}} \cap \left\{\|\zeta\|_{L^\infty(\mathcal{M})} \le \frac{\sqrt{d}}{2}\right\}\right] = 0.$$

By our previous reductions, we conclude

$$\mathbb{P}\left[\mathcal{E}_i^N \cap \bigcap_{j=0}^{i-1}\big(\mathcal{E}_j^N\big)^{\mathsf{c}}\right] \le 2e^{-cd},$$

and in particular

$$\mathbb{P}\left[\bigcup_{i=0}^{k}\mathcal{E}_k^N\right] \le (2k+1)e^{-cd}.$$

The claim is then established by taking $k$ as large as $L^q/(n\tau)$. $\qquad\square$

**Corollary B.11.** *Write $\Theta_{\mu^N}$ for the operator defined in Lemma B.9, with the parameters $\boldsymbol{\theta}$ set to the initial random network weights $\boldsymbol{\theta}_0$ and the measure set to $\mu^N$, and define for $k \in \mathbb{N}_0$*

$$\Delta_k^N = \max_{i \in \{0,1,\ldots,k\}} \left\| \Theta_i^N - \Theta \right\|_{L^\infty(\mathcal{M} \times \mathcal{M})}.$$

*There exist absolute constants $c, C, C', K, K' > 0$ such that for any $q \geq 0$ and any $d \geq K d_0 \log(n n_0 C_{\mathcal{M}})$, if*

$$\tau < \frac{1}{\left\| \boldsymbol{\Theta}_{\mu^N} \right\|_{L^2_{\mu^N}(\mathcal{M}) \to L^2_{\mu^N}(\mathcal{M})}}.$$

*and if in addition $n \geq K' L^{48+20q} d^9 \log^9 L$, then one has on an event of probability at least $1 - C'(1 + L^q/(n\tau))e^{-cd}$*

$$\Delta_{\lfloor L^q/(n\tau) \rfloor - 1}^N \leq C \log^{3/4}(L) d^{3/4} L^{4+2q/3} n^{11/12}.$$

*Proof.* Use Lemma B.10 to remove the hypothesis about boundedness of the errors from Lemma F.5, then apply this result together with a union bound. $\square$

**Lemma B.12.** *Write $\Theta$ for the operator defined in Lemma B.9, with the parameters $\boldsymbol{\theta}$ set to the initial random network weights and the measure set to $\mu^\infty$. Consider the (population) nominal error evolution $\zeta_k^\infty$, defined iteratively as*

$$\zeta_{k+1}^\infty = \zeta_k^\infty - \tau \Theta \left[ \zeta_k^\infty \right];$$
$$\zeta_0^\infty = \zeta$$

*for a step size $\tau > 0$, which satisfies*

$$\tau < \frac{1}{\left\| \boldsymbol{\Theta} \right\|_{L^2_{\mu^\infty}(\mathcal{M}) \to L^2_{\mu^\infty}(\mathcal{M})}}.$$

*Then for any $g \in L^2_{\mu^\infty}(\mathcal{M})$ and any $k$ satisfying*

$$k\tau \geq \sqrt{\frac{3e}{2}} \frac{\|g\|_{L^2_{\mu^\infty}(\mathcal{M})}}{\|\zeta\|_{L^\infty(\mathcal{M})}},$$

*we have*

$$\|\zeta_k^\infty\|_{L^2_{\mu^\infty}(\mathcal{M})} \leq \sqrt{3} \|\Theta[g] - \zeta\|_{L^2_{\mu^\infty}(\mathcal{M})} - \frac{3\|g\|_{L^2_{\mu^\infty}(\mathcal{M})}}{k\tau} \log\left( \sqrt{\frac{3}{2}} \frac{\|g\|_{L^2_{\mu^\infty}(\mathcal{M})}}{\|\zeta\|_{L^\infty(\mathcal{M})} k\tau} \right).$$

*Proof.* The dynamics satisfy the 'update equation'

$$\zeta_k^\infty = (\mathrm{Id} - \tau \Theta)^k [\zeta].$$

Because $\mathcal{M}$ is compact and $\zeta$ is a continuous function of the input, we have $\zeta \in L^\infty(\mathcal{M})$ for all values of the random weights. Because $\mu^\infty$ is a probability measure, this means $\zeta$ has finite $L^p_{\mu^\infty}(\mathcal{M})$ norm for every $p > 0$. Using the eigendecomposition of $\Theta$ as developed in Lemma B.9, we can therefore write

$$\zeta = \sum_{i=0}^\infty \langle v_i, \zeta \rangle_{L^2_{\mu^\infty}(\mathcal{M})} v_i$$

in the sense of $L^2_{\mu^\infty}(\mathcal{M})$. Because $\Theta$ and $\mathrm{Id} - \tau\Theta$ diagonalize simultaneously, we obtain

$$\|\zeta_k^\infty\|^2_{L^2_{\mu^\infty}(\mathcal{M})} = \sum_{i=1}^\infty (1 - \tau\lambda_i)^{2k} \langle v_i, \zeta \rangle^2_{L^2_{\mu^\infty}(\mathcal{M})} \leq \sum_{i=1}^\infty e^{-2k\tau\lambda_i} \langle v_i, \zeta \rangle^2_{L^2_{\mu^\infty}(\mathcal{M})},$$

where the inequality follows from the elementary estimate $1 - x \leq e^{-x}$ for $x \geq 0$ and our choice of $\tau$, which guarantees that $1 - \tau\lambda_i > 0$ for all $i \in \mathbb{N}$ so that the elementary estimate is valid after squaring. We can split this last sum into two parts: for any $\lambda \in \mathbb{R}$, we have

$$\|\zeta_k^\infty\|^2_{L^2_{\mu^\infty}(\mathcal{M})} = \sum_{i\,:\,\lambda_i \geq \lambda} e^{-2k\tau\lambda_i} \langle v_i, \zeta \rangle^2_{L^2_{\mu^\infty}(\mathcal{M})} + \sum_{i\,:\,\lambda_i < \lambda} e^{-2k\tau\lambda_i} \langle v_i, \zeta \rangle^2_{L^2_{\mu^\infty}(\mathcal{M})}.$$

Because $\boldsymbol{\Theta}$ is positive, we have further that $\lambda_i \geq 0$ for all $i$, so we can take $\lambda \geq 0$. The first sum consists of large eigenvalues: we use $\exp(-2k\tau\lambda_i) \leq \exp(-2k\tau\lambda)$ to preserve their effect, and then upper bound the remainder of the sum by the squared $L^2_{\mu^\infty}$ norm of $\zeta$. The second sum consists of small eigenvalues: we replace $\exp(-2k\tau\lambda_i) \leq 1$, and then plug in $\zeta = \boldsymbol{\Theta}[g] - (\boldsymbol{\Theta}[g] - \zeta)$ and use bilinearity, self-adjointness of $\boldsymbol{\Theta}$, and the triangle inequality to get

$$\left| \langle v_i, \zeta \rangle_{L^2_{\mu^\infty}(\mathcal{M})} \right| \leq \left| \lambda \langle v_i, g \rangle_{L^2_{\mu^\infty}(\mathcal{M})} \right| + \left| \langle v_i, \boldsymbol{\Theta}[g] - \zeta \rangle_{L^2_{\mu^\infty}(\mathcal{M})} \right|.$$

We then square both (nonnegative) sides of the inequality and use Cauchy-Schwarz to replace the squared sum with the sum of squares times a constant, obtaining

$$\|\zeta_k^\infty\|^2_{L^2_{\mu^\infty}(\mathcal{M})} \leq e^{-2k\tau\lambda}\|\zeta\|^2_{L^2_{\mu^\infty}(\mathcal{M})} + 3\lambda^2\|g\|^2_{L^2_{\mu^\infty}(\mathcal{M})} + 3\|\boldsymbol{\Theta}[g] - \zeta\|^2_{L^2_{\mu^\infty}(\mathcal{M})}$$

after re-adding indices $i$ to the sum to obtain the third residual. We will choose $\lambda \geq 0$ to minimize the sum of the first and second terms. Differentiating and setting to zero gives the critical point equation

$$\frac{2}{3}\frac{\|\zeta\|^2_{L^2_{\mu^\infty}(\mathcal{M})}(k\tau)^2}{\|g\|^2_{L^2_{\mu^\infty}(\mathcal{M})}} = (2t\lambda)e^{2k\tau\lambda},$$

which can be inverted to give the unique critical point

$$\lambda = \frac{1}{2k\tau}W\left(\frac{2}{3}\frac{\|\zeta\|^2_{L^2_{\mu^\infty}(\mathcal{M})}(k\tau)^2}{\|g\|^2_{L^2_{\mu^\infty}(\mathcal{M})}}\right),$$

where $W$ is the Lambert $W$ function, defined as the principal branch of the inverse of $z \mapsto ze^z$; we know that this critical point is a minimizer because the function of $\lambda$ we differentiated diverges as $\lambda \to \infty$. Plugging this point into the sum of the first two terms gives

$$\|\zeta_k^\infty\|^2_{L^2_{\mu^\infty}(\mathcal{M})} \leq 3\|\boldsymbol{\Theta}[g] - \zeta\|^2_{L^2_{\mu^\infty}(\mathcal{M})}$$
$$+ \frac{\|g\|^2_{L^2_{\mu^\infty}(\mathcal{M})}}{(2/3)(k\tau)^2}\left(1 + \frac{1}{2}W\left(\frac{\|\zeta\|^2_{L^2_{\mu^\infty}(\mathcal{M})}(k\tau)^2}{(3/2)\|g\|^2_{L^2_{\mu^\infty}(\mathcal{M})}}\right)\right)W\left(\frac{\|\zeta\|^2_{L^2_{\mu^\infty}(\mathcal{M})}(k\tau)^2}{(3/2)\|g\|^2_{L^2_{\mu^\infty}(\mathcal{M})}}\right).$$

For $x \geq 0$, the function $x \mapsto W(x)$ is strictly increasing, as the inverse of $y \mapsto ye^y$; by definition $W(e) = 1$; and we have the representation $W(z) + \log W(z) = \log z$ (Corless et al., 1996), whence $W(x) \leq \log x$ if $x \geq e$. Because $\mu^\infty$ is a probability measure, we have

$$\|\zeta\|^2_{L^2_{\mu^\infty}(\mathcal{M})} \leq \|\zeta\|^2_{L^\infty},$$

and therefore if

$$k\tau \geq \sqrt{\frac{3e}{2}}\frac{\|g\|_{L^2_{\mu^\infty}(\mathcal{M})}}{\|\zeta\|_{L^\infty(\mathcal{M})}},$$

we can simplify the previous bound to

$$\|\zeta_k^\infty\|^2_{L^2_{\mu^\infty}(\mathcal{M})} \leq 3\|\boldsymbol{\Theta}[g] - \zeta\|^2_{L^2_{\mu^\infty}(\mathcal{M})} + \frac{9\|g\|^2_{L^2_{\mu^\infty}(\mathcal{M})}}{4(k\tau)^2}\log^2\left(\frac{3}{2}\frac{\|g\|^2_{L^2_{\mu^\infty}(\mathcal{M})}}{\|\zeta\|^2_{L^\infty(\mathcal{M})}(k\tau)^2}\right),$$

using also properties of the logarithm. Taking square roots and using the Minkowski inequality then yields

$$\|\zeta_k^\infty\|_{L^2_{\mu^\infty}(\mathcal{M})} \leq \sqrt{3}\|\boldsymbol{\Theta}[g] - \zeta\|_{L^2_{\mu^\infty}(\mathcal{M})} - \frac{3\|g\|_{L^2_{\mu^\infty}(\mathcal{M})}}{k\tau}\log\left(\sqrt{\frac{3}{2}}\frac{\|g\|_{L^2_{\mu^\infty}(\mathcal{M})}}{\|\zeta\|_{L^\infty(\mathcal{M})}k\tau}\right),$$

where we used the previous lower bound on $k\tau$ to determine the sign that the absolute value of the logarithm takes. This gives the claim. $\qquad\square$

**Lemma B.13** (Kantorovich-Rubinstein Duality). *Let* $\mathrm{Lip}(\mathcal{M})$ *denote the class of functions* $f :$ $\mathcal{M} \to \mathbb{R}$ *such that both* $f\big|_{\mathcal{M}_\pm}$ *are Lipschitz with respect to the Riemannian distances on* $\mathcal{M}_\pm$. *For any* $d \geq 1$, *any* $0 < \delta \leq 1$ *and any* $N \geq 2\sqrt{d}/\min\{\mu^\infty(\mathcal{M}_+), \mu^\infty(\mathcal{M}_-)\}$, *one has that on an event of probability at least* $1 - 8e^{-d}$, *simultaneously for all* $f \in \mathrm{Lip}(\mathcal{M})$

$$\left| \int_{\mathcal{M}} f(\boldsymbol{x}) \, \mathrm{d}\mu^\infty(\boldsymbol{x}) - \int_{\mathcal{M}} f(\boldsymbol{x}) \, \mathrm{d}\mu^N(\boldsymbol{x}) \right|$$

$$\leq \frac{2\|f\|_{L^\infty(\mathcal{M})}\sqrt{d}}{N} + \frac{e^{14/\delta}C_{\mu^\infty,\mathcal{M}}\sqrt{d}\max_{\star \in \{+,-\}}\left\|f\big|_{\mathcal{M}_\star}\right\|_{\mathrm{Lip}}}{N^{1/(2+\delta)}},$$

*where*

$$C_{\mu^\infty,\mathcal{M}} = \frac{\mathrm{len}\,(\mathcal{M}_+)}{\mu^\infty(\mathcal{M}_+)} + \frac{\mathrm{len}\,(\mathcal{M}_-)}{\mu^\infty(\mathcal{M}_-)}.$$

*Proof.* The proof is an application of the Kantorovich-Rubinstein duality theorem for the 1-Wasserstein distance (Weed & Bach, 2019, eq. (1)), which states that for any two Borel probability measures $\mu, \nu$ on $\mathcal{M}_\pm$, one has

$$\mathcal{W}(\mu, \nu) = \sup_{\|f\|_{\mathrm{Lip}} \leq 1} \left| \int_{\mathcal{M}_\pm} f(\boldsymbol{x}) \, \mathrm{d}\mu(\boldsymbol{x}) - \int_{\mathcal{M}_\pm} f(\boldsymbol{x}) \, \mathrm{d}\nu(\boldsymbol{x}) \right|,$$

where $\mathcal{M}_\pm$ denotes either of $\mathcal{M}_+$ or $\mathcal{M}_-$, and $\|\cdot\|_{\mathrm{Lip}}$ is the minimal Lipschitz constant with respect to the Riemannian distance on $\mathcal{M}_\pm$. Therefore for any $f : \mathcal{M}_\pm \to \mathbb{R}$ Lipschitz, we have

$$\left| \int_{\mathcal{M}_\pm} f(\boldsymbol{x}) \, \mathrm{d}\mu(\boldsymbol{x}) - \int_{\mathcal{M}_\pm} f(\boldsymbol{x}) \, \mathrm{d}\nu(\boldsymbol{x}) \right| \leq \|f\|_{\mathrm{Lip}}\mathcal{W}(\mu, \nu), \tag{B.83}$$

where one checks separately the case where $\|f\|_{\mathrm{Lip}} = 0$ to see that this bound holds there as well. To go from (B.83) to the desired conclusion, we need to pass from the measures $\mu^\infty$ and $\mu^N$, both supported on $\mathcal{M}$, to measures $\mu_\pm^\star$ (with $\star \in \{N, \infty\}$), supported on the manifolds $\mathcal{M}_\pm$ (which we will define in detail below); the challenge here is that the number of 'hits' of each manifold $\mathcal{M}_\pm$ that show up in the finite sample measure $\mu^N$ is a random variable, which requires a small detour to control. Let us define random variables $N_+$, $N_-$ by

$$N_+ = N\mu^N(\mathcal{M}_+); \qquad N_- = N\mu^N(\mathcal{M}_-),$$

so that $N_\pm$ have support in $\{0, 1, \ldots, N\}$, and $N_+ + N_- = N$. Define in addition

$$p_+ = \mu^\infty(\mathcal{M}_+); \qquad p_- = \mu^\infty(\mathcal{M}_-),$$

which represent the degree of imbalance between the positive and negative classes in the data. By definition of the i.i.d. sample, we have that $N_+ \sim \mathrm{Binom}(N, p_+)$. Using $N_+$ and $N_-$, we can define 'conditional' finite sample measures $\mu_+^N$ and $\mu_-^N$ by

$$\mu_+^N = \frac{1}{\max\{1, N_+\}} \sum_{i \in [N]\,:\,\boldsymbol{x}_i \in \mathcal{M}_+} \delta_{\{\boldsymbol{x}_i\}}; \qquad \mu_-^N = \frac{1}{\max\{1, N_-\}} \sum_{i \in [N]\,:\,\boldsymbol{x}_i \in \mathcal{M}_-} \delta_{\{\boldsymbol{x}_i\}},$$

so that $(N_+/N)\mu_+^N + (N_-/N)\mu_-^N = \mu^N$,[8] and $\mu_+^N$ and $\mu_-^N$ are both probability measures except when $N_+ \in \{0, N\}$, in which case exactly one is a probability measure. By the triangle inequality, we have for any continuous $f : \mathcal{M} \to \mathbb{R}$

$$\left| \int_{\mathcal{M}} f(\boldsymbol{x}) \, \mathrm{d}\mu^\infty(\boldsymbol{x}) - \int_{\mathcal{M}} f(\boldsymbol{x}) \, \mathrm{d}\mu^N(\boldsymbol{x}) \right|$$

$$\leq \sum_{\star \in \{+,-\}} \left| p_\star \int_{\mathcal{M}_\star} f(\boldsymbol{x}) \frac{\mathrm{d}\mu_\star^\infty(\boldsymbol{x})}{p_\star} - \frac{N_\star}{N} \int_{\mathcal{M}_\star} f(\boldsymbol{x}) \, \mathrm{d}\mu_\star^N(\boldsymbol{x}) \right|$$

$$\leq \sum_{\star \in \{+,-\}} \|f\|_{L^\infty(\mathcal{M})} \left| \frac{N_\star}{N} - p_\star \right| + \left| \int_{\mathcal{M}_\star} f(\boldsymbol{x}) \frac{\mathrm{d}\mu_\star^\infty(\boldsymbol{x})}{p_\star} - \int_{\mathcal{M}_\star} f(\boldsymbol{x}) \, \mathrm{d}\mu_\star^N(\boldsymbol{x}) \right|. \tag{B.84}$$

---

[8]Here we treat the empty sum as the appropriate 'zero element' of the space of finite signed Borel measures on $\mathcal{M}_\pm$, namely the trivial measure that assigns zero to every Borel subset of $\mathcal{M}_\pm$.

By Lemma G.1, we have

$$\mathbb{P}\left[\left|\frac{N_\star}{N} - p_\star\right| \le \frac{\sqrt{d}}{N}\right] \ge 1 - 2e^{-2d}. \tag{B.85}$$

Using that $N - N_+ = N_-$ and $1 - p_+ = p_-$, the bound (B.85) implies if $N \ge 2\sqrt{d}/\min\{p_+, p_-\}$

$$\mathbb{P}\left[\frac{p_\star}{2} \le \frac{N_\star}{N} \le \frac{1 - p_\star}{2}\right] \ge 1 - 2e^{-2d}. \tag{B.86}$$

Now fix an arbitrary $f \in \mathrm{Lip}(\mathcal{M})$. For either $\star \in \{+, -\}$, we can write

$$\int_{\mathcal{M}_\star} f(\boldsymbol{x}) \, \mathrm{d}\mu_\star^N(\boldsymbol{x}) = \frac{1}{\max\{1, N_\star\}} \sum_{i\,:\,\boldsymbol{x}_i \in \mathcal{M}_\star} f(\boldsymbol{x}_i) = \frac{1}{\max\{1, \sum_{i=1}^N \mathbb{1}_{\boldsymbol{x}_i \in \mathcal{M}_\star}\}} \sum_{i=1}^N \mathbb{1}_{\boldsymbol{x}_i \in \mathcal{M}_\star} f(\boldsymbol{x}_i),$$

and since $\mathcal{M}_+$ and $\mathcal{M}_-$ are separated by a positive distance $\Delta > 0$, we have that $\boldsymbol{x}_i \mapsto \mathbb{1}_{\boldsymbol{x}_i \in \mathcal{M}_\star}$ are continuous functions on $\mathcal{M}$. Since $f$ is continuous on $\mathcal{M}$ by the same reasoning and the fact that $\mathcal{M}$ is compact, it follows that the functions $(\boldsymbol{x}_1, \ldots, \boldsymbol{x}_N) \mapsto \int_{\mathcal{M}_\star} f(\boldsymbol{x}) \, \mathrm{d}\mu_\star^N(\boldsymbol{x})$ are continuous on $\mathcal{M} \times \cdots \times \mathcal{M}$ as well, and in particular for any $t > 0$ the sets

$$\left\{\left|\int_{\mathcal{M}_\star} f(\boldsymbol{x})\frac{\mathrm{d}\mu_\star^\infty(\boldsymbol{x})}{p_\star} - \int_{\mathcal{M}_\star} f(\boldsymbol{x}) \, \mathrm{d}\mu_\star^N(\boldsymbol{x})\right| > t\right\}$$

are open in $\mathcal{M}$, and so is their union over all $f \in \mathrm{Lip}(\mathcal{M})$. By conditioning, we can then apply (B.86) to write

$$\mathbb{P}\left[\bigcup_{f \in \mathrm{Lip}(\mathcal{M})}\left\{\left|\int_{\mathcal{M}_\star} f(\boldsymbol{x})\frac{\mathrm{d}\mu_\star^\infty(\boldsymbol{x})}{p_\star} - \int_{\mathcal{M}_\star} f(\boldsymbol{x}) \, \mathrm{d}\mu_\star^N(\boldsymbol{x})\right| > t\right\}\right]$$

$$\le 2e^{-2d} + \sum_{k=\lfloor Np_\star/2\rfloor}^{\lceil N(1-p_\star)/2\rceil} \mathbb{P}\left[\bigcup_{f \in \mathrm{Lip}(\mathcal{M})}\left\{\left|\begin{array}{c}\int_{\mathcal{M}_\star} f(\boldsymbol{x})\frac{\mathrm{d}\mu_\star^\infty(\boldsymbol{x})}{p_\star}\\ -\int_{\mathcal{M}_\star} f(\boldsymbol{x}) \, \mathrm{d}\mu_\star^N(\boldsymbol{x})\end{array}\right| > t\right\}\;\middle|\; N_\star = k\right]\mathbb{P}[N_\star = k]. \tag{B.87}$$

Conditioned on $\{N_\star = k\}$ with $0 < k < N$, the measure $\mu_\star^N$ is distributed as an empirical measure of sample size $k$ from the probability measure $\mu_\star^\infty/p_\star$ supported on $\mathcal{M}_\star$. For $\lfloor Np_\star/2\rfloor \le k \le \lceil N(1 - p_\star)/2\rceil$, any $\delta > 0$ and any $d \ge 1$ we have for both possible values of $\star$

$$\frac{\sqrt{d}e^{14/\delta}\,\mathrm{len}(\mathcal{M}_\star)}{k^{1/(2+\delta)}} \le \frac{\sqrt{d}e^{14/\delta}\,\mathrm{len}(\mathcal{M}_\star)}{\left(\left\lfloor\frac{Np_\star}{2}\right\rfloor\right)^{1/(2+\delta)}}$$

$$\le \frac{\sqrt{2d}e^{14/\delta}\,\mathrm{len}(\mathcal{M}_\star)}{(Np_\star)^{1/(2+\delta)}},$$

and so an application of Lemma B.16 thus gives for any $0 < \delta \le 1$ and any $d \ge 2$

$$\mathbb{P}\left[\mathcal{W}\left(\frac{\mathrm{d}\mu_\star^\infty(\boldsymbol{x})}{p_\star}, \mathrm{d}\mu_\star^N\right) > \frac{\sqrt{d}e^{14/\delta}\,\mathrm{len}(\mathcal{M}_\star)}{(Np_\star)^{1/(2+\delta)}}\;\middle|\; N_\star = k\right] \le e^{-d}.$$

Combining this last bound with (B.83) and (B.87) gives

$$\mathbb{P}\left[\bigcup_{f \in \mathrm{Lip}(\mathcal{M})}\left\{\begin{array}{c}\left|\int_{\mathcal{M}_\star} f(\boldsymbol{x})\frac{\mathrm{d}\mu_\star^\infty(\boldsymbol{x})}{p_\star} - \int_{\mathcal{M}_\star} f(\boldsymbol{x}) \, \mathrm{d}\mu_\star^N(\boldsymbol{x})\right|\\ > \frac{\sqrt{d}e^{14/\delta}\left\|f\right|_{\mathcal{M}_\star}\right\|_{\mathrm{Lip}}\,\mathrm{len}(\mathcal{M}_\star)}{N^{1/(2+\delta)}\,p_\star}\end{array}\right\}\right] \le 3e^{-d}.$$

where we used $\max\{p_+, p_-\} \le 1$ to remove the exponent of $1/(2 + \delta)$ on these terms. Taking a max over the Lipschitz constants and combining this bound with (B.84) and (B.85) and a union bound, we obtain

$$\mathbb{P}\left[\bigcup_{f \in \mathrm{Lip}(\mathcal{M})}\left\{\begin{array}{c}\left|\int_{\mathcal{M}} f(\boldsymbol{x}) \, \mathrm{d}\mu^\infty(\boldsymbol{x}) - \int_{\mathcal{M}} f(\boldsymbol{x}) \, \mathrm{d}\mu^N(\boldsymbol{x})\right|\\ > \frac{2\|f\|_{L^\infty(\mathcal{M})}\sqrt{d}}{N}\\ + \frac{e^{14/\delta}C_{\mu^\infty,\mathcal{M}}\sqrt{d}\max_{\star \in \{+,-\}}\left\|f\right|_{\mathcal{M}_\star}\right\|_{\mathrm{Lip}}}{N^{1/(2+\delta)}}\end{array}\right\}\right] \le 8e^{-d},$$

where the constant is defined as in the statement of the lemma. $\qquad\square$

**Lemma B.14.** *Let $d_0 = 1$. There is an absolute constant $C > 0$ such that for any function $f :$ $\mathcal{M} \to \mathbb{R}$ with $f\big|_{\mathcal{M}_\pm}$ Lipschitz with respect to the Riemannian distances on $\mathcal{M}_\pm$, one has*

$$\|f\|_{L^\infty} \le C \max\left\{ \begin{array}{c} \dfrac{\rho_{\max}^{1/2}\|f\|_{L_{\mu^\infty}^2(\mathcal{M})}}{\rho_{\min}^{1/2}(\min\{\mu^\infty(\mathcal{M}_+),\mu^\infty(\mathcal{M}_-)\})^{1/2}}, \\ \dfrac{\|f\|_{L_{\mu^\infty}^2(\mathcal{M})}^{2/3}\max_{\star\in\{+,-\}}\left\|f\big|_{\mathcal{M}_\star}\right\|_{\mathrm{Lip}}^{1/3}}{\rho_{\min}^{1/3}} \end{array} \right\}.$$

*Proof.* For any $T > 0$ and a nonconstant Lipschitz function $f : [0, T] \to \mathbb{R}$, we will establish the inequality

$$\|f\|_{L^\infty} \le C \max\left\{ \frac{\|f\|_{L^2}}{\sqrt{T}}, \|f\|_{L^2}^{2/3}\|f\|_{\mathrm{Lip}}^{1/3} \right\}, \tag{B.88}$$

where the constant $C > 0$ is absolute. We can use this result to establish the claim. We start by writing

$$\|f\|_{L^\infty} = \max_{\star\in\{+,-\}} \left\|f\big|_{\mathcal{M}_\star}\right\|_{L^\infty},$$

and for $\star \in \{+, -\}$, we have

$$\left\|f\big|_{\mathcal{M}_\star}\right\|_{L^\infty} = \|f \circ \gamma_\star\|_{L^\infty}, \tag{B.89}$$

where $\gamma_\star : [0, \mathrm{len}(\mathcal{M}_\star)] \to \mathcal{M}_\star$ are the smooth unit-speed curves parameterized with respect to arc length parameterizing the manifolds. Similarly, the curves' parameterization with respect to arc length implies

$$\|f \circ \gamma_\star\|_{\mathrm{Lip}} \le \left\|f\big|_{\mathcal{M}_\star}\right\|_{\mathrm{Lip}}. \tag{B.90}$$

Applying (B.88) with (B.89) and (B.90), we obtain

$$\left\|f\big|_{\mathcal{M}_\star}\right\|_{L^\infty} \le C \max\left\{ \frac{\|f \circ \gamma_\star\|_{L^2}}{\sqrt{\mathrm{len}(\mathcal{M}_\star)}}, \|f \circ \gamma_\star\|_{L^2}^{2/3}\left\|f\big|_{\mathcal{M}_\star}\right\|_{\mathrm{Lip}}^{1/3} \right\}.$$

For the first term in the max, we have

$$\frac{\|f \circ \gamma_\star\|_{L^2}^2}{\mathrm{len}(\mathcal{M}_\star)}$$

$$= \left| \frac{1}{\mathrm{len}(\mathcal{M}_\star)} \int_0^{\mathrm{len}(\mathcal{M}_\star)} f \circ \gamma_\star(t)^2 \, \mathrm{d}t \right|$$

$$\le \left| \int_0^{\mathrm{len}(\mathcal{M}_\star)} f \circ \gamma_\star(t)^2 \rho_\star \circ \gamma_\star(t) \frac{\rho_\star \circ \gamma_\star(t) - \frac{1}{\mathrm{len}(\mathcal{M}_\star)}}{\rho_\star \circ \gamma_\star(t)} \, \mathrm{d}t \right|$$

$$+ \left| \int_0^{\mathrm{len}(\mathcal{M}_\star)} f \circ \gamma_\star(t)^2 \rho_\star \circ \gamma_\star(t) \, \mathrm{d}t \right|$$

using the triangle inequality. For the second term in the last bound, we note that

$$\left| \int_0^{\mathrm{len}(\mathcal{M}_\star)} f \circ \gamma_\star(t)^2 \rho_\star \circ \gamma_\star(t) \, \mathrm{d}t \right|$$

$$\le \left| \int_0^{\mathrm{len}(\mathcal{M}_+)} f \circ \gamma_+(t)^2 \rho_+ \circ \gamma_+(t) \, \mathrm{d}t \right| + \left| \int_0^{\mathrm{len}(\mathcal{M}_-)} f \circ \gamma_-(t)^2 \rho_- \circ \gamma_-(t) \, \mathrm{d}t \right|$$

$$\le \|f\|_{L_{\mu^\infty}^2(\mathcal{M})}^2, \tag{B.91}$$

and for the first term, we have

$$\max_{t \in [0, \operatorname{len}(\mathcal{M}_\star)]} \left| \frac{\rho_\star \circ \boldsymbol{\gamma}_\star(t) - \frac{1}{\operatorname{len}(\mathcal{M}_\star)}}{\rho_\star \circ \boldsymbol{\gamma}_\star(t)} \right|$$

$$\leq \max_{t \in [0, \operatorname{len}(\mathcal{M}_\star)]} \frac{\left| \rho_\star \circ \boldsymbol{\gamma}_\star(t) - \frac{\rho_\star \circ \boldsymbol{\gamma}_\star(t)}{\mu^\infty(\mathcal{M}_\star)} \right| + \left| \frac{\rho_\star \circ \boldsymbol{\gamma}_\star(t)}{\mu^\infty(\mathcal{M}_\star)} - \frac{1}{\operatorname{len}(\mathcal{M}_\star)} \right|}{\rho_\star \circ \boldsymbol{\gamma}_\star(t)}$$

$$\leq \frac{1 - \mu^\infty(\mathcal{M}_\star)}{\mu^\infty(\mathcal{M}_\star)} + \frac{\rho_{\max}}{\mu^\infty(\mathcal{M}_\star)\rho_{\min}}$$

$$\leq \frac{2\rho_{\max}}{\mu^\infty(\mathcal{M}_\star)\rho_{\min}}, \tag{B.92}$$

where in the first inequality we used the triangle inequality, and for the second we used that $\rho_\star \circ \boldsymbol{\gamma}_\star$ integrates to $\mu^\infty(\mathcal{M}_\star)$ over $[0, \operatorname{len}(\mathcal{M}_\star)]$, which implies that there exists at least one $t \in [0, \operatorname{len}(\mathcal{M}_\star)]$ at which $\rho_\star \circ \boldsymbol{\gamma}_\star(t) \geq \mu^\infty(\mathcal{M}_\star)/\operatorname{len}(\mathcal{M}_\star)$, so that the maximum of the difference in the second term on the RHS of the first inequality is bounded by the maximum of the density term. Thus, by Hölder's inequality and (B.91) and (B.92), we have

$$\left| \int_0^{\operatorname{len}(\mathcal{M}_\star)} f \circ \boldsymbol{\gamma}_\star(t)^2 \rho_\star \circ \boldsymbol{\gamma}_\star(t) \frac{\rho_\star \circ \boldsymbol{\gamma}_\star(t) - \frac{1}{\operatorname{len}(\mathcal{M}_\star)}}{\rho_\star \circ \boldsymbol{\gamma}_\star(t)} \, \mathrm{d}t \right|$$

$$\leq \frac{3\rho_{\max}}{\rho_{\min} \min \{\mu^\infty(\mathcal{M}_+), \mu^\infty(\mathcal{M}_-)\}} \|f\|_{L^2_{\mu^\infty}(\mathcal{M})}^2$$

Similarly, for the second term in the max, we have

$$\|f \circ \boldsymbol{\gamma}_\star\|_{L^2}^{2/3}$$

$$= \left( \int_0^{\operatorname{len}(\mathcal{M}_\star)} f \circ \boldsymbol{\gamma}_\star(t)^2 \, \mathrm{d}t \right)^{1/3}$$

$$\leq \left( \int_0^{\operatorname{len}(\mathcal{M}_+)} f \circ \boldsymbol{\gamma}_+(t)^2 \, \mathrm{d}t + \int_0^{\operatorname{len}(\mathcal{M}_-)} f \circ \boldsymbol{\gamma}_-(t)^2 \, \mathrm{d}t \right)^{1/3}$$

$$\leq \frac{1}{\rho_{\min}^{1/3}} \left( \int_0^{\operatorname{len}(\mathcal{M}_+)} f \circ \boldsymbol{\gamma}_+(t)^2 \rho_+ \circ \boldsymbol{\gamma}_+(t) \, \mathrm{d}t + \int_0^{\operatorname{len}(\mathcal{M}_-)} f \circ \boldsymbol{\gamma}_-(t)^2 \rho_- \circ \boldsymbol{\gamma}_-(t) \, \mathrm{d}t \right)^{1/3}$$

$$\leq \frac{\|f\|_{L^2_{\mu^\infty}(\mathcal{M})}^{2/3}}{\rho_{\min}^{1/3}}.$$

Thus, we have

$$\left\| f|_{\mathcal{M}_\star} \right\|_{L^\infty} \leq C \max \left\{ \frac{\rho_{\max}^{1/2} \|f\|_{L^2_{\mu^\infty}(\mathcal{M})}}{\rho_{\min}^{1/2} (\min \{\mu^\infty(\mathcal{M}_+), \mu^\infty(\mathcal{M}_-)\})^{1/2}}, \frac{\|f\|_{L^2_{\mu^\infty}(\mathcal{M})}^{2/3} \left\| f|_{\mathcal{M}_\star} \right\|_{\operatorname{Lip}}^{1/3}}{\rho_{\min}^{1/3}} \right\},$$

and taking a maximum over $\star \in \{+, -\}$ establishes the claim.

To prove (B.88), consider first the trivial case where $\|f\|_{L^\infty} = 0$: here the LHS and RHS of (B.88) are identical, and the proof is immediate. When $\|f\|_{L^\infty} > 0$, the Weierstrass theorem implies that there exists $t \in [0, T]$ such that $|f(t)| = \|f\|_{L^\infty}$; we consider the case $\operatorname{sign}(f(t)) > 0$. For any $t' \in [0, T]$, we can write by the Lipschitz property

$$f(t') \geq \|f\|_{L^\infty} - \|f\|_{\operatorname{Lip}} |t - t'|,$$

and the RHS of the previous bound is nonnegative on the intersection of the interval $[t - \|f\|_{L^\infty} \|f\|_{\operatorname{Lip}}^{-1}, t + \|f\|_{L^\infty} \|f\|_{\operatorname{Lip}}^{-1}]$ with the domain $[0, T]$ (with standard extended-valued arithmetic

conventions when $\|f\|_{\mathrm{Lip}} = 0$). This gives the bound

$$\|f\|_{L^2}^2 \geq \int_{\max\left\{t - \frac{\|f\|_{L^\infty}}{\|f\|_{\mathrm{Lip}}}, 0\right\}}^{\min\left\{t + \frac{\|f\|_{L^\infty}}{\|f\|_{\mathrm{Lip}}}, T\right\}} \left(\|f\|_{L^\infty} - \|f\|_{\mathrm{Lip}}|t - t'|\right)^2 \mathrm{d}t'$$

$$= \int_{\max\left\{-\frac{\|f\|_{L^\infty}}{\|f\|_{\mathrm{Lip}}}, -t\right\}}^{\min\left\{\frac{\|f\|_{L^\infty}}{\|f\|_{\mathrm{Lip}}}, T-t\right\}} \left(\|f\|_{L^\infty} - \|f\|_{\mathrm{Lip}}|t'|\right)^2 \mathrm{d}t',$$

where the second line follows from the changes of variables $t' \mapsto t' + t$. The integrand on the RHS of the second line in the previous bound is even-symmetric, and $\max\left\{-\frac{\|f\|_{L^\infty}}{\|f\|_{\mathrm{Lip}}}, -t\right\} = -\min\left\{\frac{\|f\|_{L^\infty}}{\|f\|_{\mathrm{Lip}}}, t\right\}$, so we can discard one side of the interval of integration to get

$$\int_{\max\left\{-\frac{\|f\|_{L^\infty}}{\|f\|_{\mathrm{Lip}}}, -t\right\}}^{\min\left\{\frac{\|f\|_{L^\infty}}{\|f\|_{\mathrm{Lip}}}, T-t\right\}} \left(\|f\|_{L^\infty} - \|f\|_{\mathrm{Lip}}|t'|\right)^2 \mathrm{d}t' \tag{B.93}$$

$$\geq \int_0^{\min\left\{\frac{\|f\|_{L^\infty}}{\|f\|_{\mathrm{Lip}}}, \max\{t, T-t\}\right\}} \left(\|f\|_{L^\infty} - \|f\|_{\mathrm{Lip}}t'\right)^2 \mathrm{d}t'. \tag{B.94}$$

We proceed analyzing two distinct cases. First, if $\|f\|_{L^\infty} \leq \max\{t, T - t\}\|f\|_{\mathrm{Lip}}$, then we must have $\|f\|_{\mathrm{Lip}} > 0$; integrating the RHS of (B.94), we obtain

$$\|f\|_{L^2}^2 \geq \frac{\|f\|_{L^\infty}^3}{3\|f\|_{\mathrm{Lip}}},$$

or equivalently

$$\|f\|_{L^\infty} \leq 3^{1/3}\|f\|_{L^2}^{2/3}\|f\|_{\mathrm{Lip}}^{1/3}. \tag{B.95}$$

Next, we consider the case $\|f\|_{L^\infty} > \max\{t, T - t\}\|f\|_{\mathrm{Lip}}$. We split on two sub-cases: when $\|f\|_{\mathrm{Lip}} = 0$, integrating (B.94) gives

$$\|f\|_{L^2}^2 \geq \|f\|_{L^\infty}^2 \max\{t, T - t\} \geq \frac{T\|f\|_{L^\infty}^2}{2}, \tag{B.96}$$

where we used $\max\{t, T - t\} \geq T/2$. When $\|f\|_{\mathrm{Lip}} > 0$, integrating (B.94) gives

$$\|f\|_{L^2}^2 \geq \frac{1}{3\|f\|_{\mathrm{Lip}}} \left(\|f\|_{L^\infty}^3 - (\|f\|_{L^\infty} - \|f\|_{\mathrm{Lip}} \max\{t, T - t\})^3\right)$$

$$= \frac{\max\{t, T - t\}}{3} \sum_{k=0}^{2} \|f\|_{L^\infty}^{2-k} (\|f\|_{L^\infty} - \|f\|_{\mathrm{Lip}} \max\{t, T - t\})^k$$

$$\geq \frac{T\|f\|_{L^\infty}^2}{6}, \tag{B.97}$$

where the second line uses a standard algebraic identity, and the third line uses $\max\{t, T - t\} \geq T/2$ together with the definition of the case to get that $\|f\|_{L^\infty} - \|f\|_{\mathrm{Lip}} \max\{t, T - t\} > 0$ in order to discard all but the $k = 0$ summand. Combining (B.97) and (B.96), we obtain for this case

$$\|f\|_{L^\infty} \leq \frac{\sqrt{6}\|f\|_{L^2}}{\sqrt{T}}, \tag{B.98}$$

and combining (B.95) and (B.98) gives unconditionally

$$\|f\|_{L^\infty} \leq \max\left\{\frac{\sqrt{6}\|f\|_{L^2}}{\sqrt{T}}, 3^{1/3}\|f\|_{L^2}^{2/3}\|f\|_{\mathrm{Lip}}^{1/3}\right\},$$

which establishes (B.88). For the case $\mathrm{sign}(f(t)) < 0$, apply the preceding argument to $-f$ to conclude. See (Brezis, 2011, Exercise 8.15) for a sketch of a proof that leads to more general versions of (B.88).

$\square$

**Lemma B.15.** *For any $p \in \mathbb{N}$, if $C \geq (4p)^{4p}$, then one has*

$$n \geq C \log^p n \quad \text{if} \quad n \geq 2^p C \log^p (2^p C).$$

*Proof.* We first give a proof for $p = 1$, then build off this proof for the general case. Consider the function $f(x) = cx - \log x$. We have $f'(x) = c - 1/x$, which is nonnegative for every $x \geq 1/c$, so in particular $f$ is increasing under this condition. By concavity of the logarithm, we have $\log x \leq \log(2/c) + (c/2)(x - 2/c)$, whence

$$f(x) \geq 1 + cx/2 - \log(2/c).$$

The RHS of this bound is equal to zero at $x = (2/c)(\log(2/c) - 1)$, and

$$\frac{2}{c} \left( \log \left( \frac{2}{c} \right) - 1 \right) \geq \frac{1}{c} \quad \Longleftrightarrow \quad c \leq 2e^{-3/2}.$$

In particular, we have $f(x) \geq 0$ for every $x \geq (2/c) \log(2/c)$. Rearranging this bound, we can assert the desired conclusion that if $C \geq 3$, then $n \geq C \log n$ for every $n \geq 2C \log 2C$. Equivalently, we have for all such $n$ that $Cn^{-1} \log n \leq 1$. Next, we consider the case of $p > 1$. We will show

$$C \frac{\log^p n}{n} \leq 1$$

under suitable conditions. Let us consider the choice $n = KC \log^p KC$, where $K > 0$ is a constant we will specify below. Consider the function $f(x) = Cx^{-1} \log^p x$, which satisfies

$$f'(x) = C \frac{\log^{p-1}(x)(p - \log^{p-1}(x))}{x^2}.$$

In particular, $f$ is decreasing as soon as $p \leq \log^{p-1}(x)$. Now, we can calculate

$$f\left(KC \log^p KC\right) = \frac{1}{K} \left( 1 + \frac{p \log \log KC}{\log KC} \right)^p,$$

and by our result for the case $p = 1$, we have for all $p \geq 2$

$$\frac{p \log \log KC}{\log KC} \leq 1 \quad \text{if} \quad \log KC \geq 4p \log 4p.$$

This condition is satisfied for $KC \geq (4p)^{4p}$, so if we set $K = 2^p$, we obtain the above conclusion when $C \geq (4p)^{4p}$. Under these conditions, we then get

$$f(KC \log^p KC) \leq 1.$$

Similarly, we have $\log^{p-1}(KC \log KC) \geq \log^{p-1}((4p)^{4p}) = (4p)^{p-1} \log^{p-1}(4p)$, which is larger than $p$ because $4p \geq e$. It follows that $f(x) \leq 1$ for every $x \geq KC \log KC$, which completes the proof. $\square$

**Lemma B.16** (Concentration of Empirical Measure in Wasserstein Distance (Weed & Bach, 2019))**.** *Let $d_0 = 1$. For either $\star \in \{+, -\}$, let $\mu$ be a Borel probability measure on $\mathcal{M}_\star$, and write $\mu^N$ for the empirical measure corresponding to $N$ i.i.d. samples from $\mu$. Then for any $d \geq 1$ and any $0 < \delta \leq 1$, one has*

$$\mathbb{P}\left[ \mathcal{W}(\mu, \mu^N) \leq \frac{\sqrt{d} e^{14/\delta} \operatorname{len}(\mathcal{M}_\star)}{N^{1/(2+\delta)}} \right] \geq 1 - e^{-2d},$$

*where the 1-Wasserstein distance is taken with respect to the Riemannian distance.*

*Proof.* The proof is a direct application of the results of (Weed & Bach, 2019) on concentration of empirical measures in Wasserstein distance. For the duration of the proof, we will work on the metric space
$(\mathcal{M}_\star, \operatorname{len}(\mathcal{M}_\star)^{-1} \operatorname{dist}_{\mathcal{M}_\star}(\cdot))$, i.e., the same metric space scaled to have unit diameter; we will then obtain the result in terms of the unscaled metric by the definition of the 1-Wasserstein distance.

Because $d_0 = 1$ and $\mathcal{M}_\star$ can be given as a unit-speed curve parameterized with respect to arc length, we have for any Borel $S \subset [0, 1]$ and any $\varepsilon > 0$

$$\mathcal{N}_\varepsilon(S) \leq \frac{1}{\varepsilon},$$

where $\mathcal{N}_\varepsilon(S)$ denotes the $\varepsilon$-covering number of $S$ by closed balls in the rescaled metric. Following the notation of (Weed & Bach, 2019, §4.1), we then obtain for any $s > 2$

$$d_\varepsilon(\mu, \varepsilon^{s/(s-2)}) = \frac{\log\inf\{\mathcal{N}_\varepsilon(S) \mid \mu(S) \geq 1 - \varepsilon^{s/(s-2)}\}}{-\log\varepsilon} \leq 1.$$

Invoking (Weed & Bach, 2019, Proposition 5), we obtain after some simplifications of the constants that for any $0 < \delta \leq 1$ (putting $s = \delta + 2$ in the previous estimates)

$$\mathbb{E}\big[\mathcal{W}\big(\mu, \mu^N\big)\big] \leq 3^{11/\delta} N^{-1/(2+\delta)} + 3^6 N^{-1/2} \leq e^{14/\delta} N^{-1/(2+\delta)},$$

where the final inequality worst-cases constants for convenience. Using (Weed & Bach, 2019, Proposition 20), we have

$$\mathbb{P}\left[\mathcal{W}(\mu, \mu^N) + \mathbb{E}\big[\mathcal{W}(\mu, \mu^N)\big] \geq \sqrt{\frac{d}{N}}\right] \leq e^{-2d},$$

and hence

$$\mathbb{P}\left[\mathcal{W}(\mu, \mu^N) \geq \frac{\sqrt{d} e^{14/\delta}}{N^{1/(2+\delta)}}\right] \leq \mathbb{P}\left[\mathcal{W}(\mu, \mu^N) \geq \frac{e^{14/\delta}}{N^{1/(2+\delta)}} + \sqrt{\frac{d}{N}}\right]$$

$$\leq \mathbb{P}\left[\mathcal{W}(\mu, \mu^N) + \mathbb{E}\big[\mathcal{W}(\mu, \mu^N)\big] \geq \sqrt{\frac{d}{N}}\right] \leq e^{-2d}$$

if $d \geq 1$. $\qquad\square$

**Lemma B.17.** *Let $n, m \in \mathbb{N}$. Let $\boldsymbol{F} : \mathbb{R}^n \to \mathbb{R}^m$ be 1-nonnegatively homogeneous, and suppose there exist $M, L \geq 0$ such that*

1. $\big\|\|\boldsymbol{F}\big|_{\mathbb{S}^{n-1}}\|_2\big\|_{L^\infty} \leq M$;

2. $\boldsymbol{F}\big|_{\mathbb{S}^{n-1}}$ *is $L$-Lipschitz.*

*Then for any $\boldsymbol{x}, \boldsymbol{x}' \in \mathbb{R}^n$, one has*
$$\|\boldsymbol{F}(\boldsymbol{x}) - \boldsymbol{F}(\boldsymbol{x}')\|_2 \leq (2L + M)\|\boldsymbol{x} - \boldsymbol{x}'\|_2,$$
*so that $\boldsymbol{F}$ is $(2L + M)$-Lipschitz.*

*Proof.* For any numbers $a, b \geq 0$ and any $\boldsymbol{u}, \boldsymbol{v} \in \mathbb{R}^m$, one has by the triangle inequality
$$\|a\boldsymbol{u} - b\boldsymbol{v}\|_2 \leq \min\{a\|\boldsymbol{u} - \boldsymbol{v}\|_2 + |a - b|\|\boldsymbol{v}\|_2, b\|\boldsymbol{u} - \boldsymbol{v}\|_2 + |a - b|\|\boldsymbol{u}\|_2\}.$$
Using an elementary property of the min and the max, we thus have
$$\|a\boldsymbol{u} - b\boldsymbol{v}\|_2 \leq \min\{a, b\}\|\boldsymbol{u} - \boldsymbol{v}\|_2 + \max\{\|\boldsymbol{u}\|_2, \|\boldsymbol{v}\|_2\}|a - b|. \tag{B.99}$$
Now we proceed to show the claim. Noting that the case where both $\boldsymbol{x}, \boldsymbol{x}'$ are zero is trivial, first consider the case where $\boldsymbol{x}$ is nonzero and $\boldsymbol{x}'$ is zero. By nonnegative homogeneity, it suffices to proceed as

$$\|\boldsymbol{F}(\boldsymbol{x}) - \boldsymbol{F}(\boldsymbol{x}')\|_2 = \|\boldsymbol{F}(\boldsymbol{x})\|_2 = \|\boldsymbol{x}\|_2 \left\|\boldsymbol{F}\left(\frac{\boldsymbol{x}}{\|\boldsymbol{x}\|_2}\right)\right\|_2 \leq M\|\boldsymbol{x}\|_2 = M\|\boldsymbol{x} - \boldsymbol{x}'\|_2$$

to conclude; for the inequality we used the boundedness assumption on $\boldsymbol{F}$. Now fix $\boldsymbol{x}, \boldsymbol{x}' \in \mathbb{R}^n$ nonzero. The inequality (B.99) can be applied to get

$$\|\boldsymbol{F}(\boldsymbol{x}) - \boldsymbol{F}(\boldsymbol{x}')\|_2 = \left\|\|\boldsymbol{x}\|_2 \boldsymbol{F}\left(\frac{\boldsymbol{x}}{\|\boldsymbol{x}\|_2}\right) - \|\boldsymbol{x}'\|_2 \boldsymbol{F}\left(\frac{\boldsymbol{x}'}{\|\boldsymbol{x}'\|_2}\right)\right\|_2$$

$$\leq \min\{\|\boldsymbol{x}\|_2, \|\boldsymbol{x}'\|_2\}\left\|\boldsymbol{F}\left(\frac{\boldsymbol{x}}{\|\boldsymbol{x}\|_2}\right) - \boldsymbol{F}\left(\frac{\boldsymbol{x}'}{\|\boldsymbol{x}'\|_2}\right)\right\|_2$$

$$+ \max\left\{\left\|\boldsymbol{F}\left(\frac{\boldsymbol{x}}{\|\boldsymbol{x}\|_2}\right)\right\|_2, \left\|\boldsymbol{F}\left(\frac{\boldsymbol{x}'}{\|\boldsymbol{x}'\|_2}\right)\right\|_2\right\}\|\boldsymbol{x} - \boldsymbol{x}'\|_2,$$

where in the inequality we also applied the $\ell^2$ triangle inequality. Using the assumed properties of $\boldsymbol{F}$, we thus have

$$\|\boldsymbol{F}(\boldsymbol{x}) - \boldsymbol{F}(\boldsymbol{x}')\|_2 \le L \min\{\|\boldsymbol{x}\|_2, \|\boldsymbol{x}'\|_2\} \left\| \frac{\boldsymbol{x}}{\|\boldsymbol{x}\|_2} - \frac{\boldsymbol{x}'}{\|\boldsymbol{x}'\|_2} \right\|_2 + M\|\boldsymbol{x} - \boldsymbol{x}'\|_2.$$

By a classical inequality (e.g. proved in (E.15)), one has

$$\left\| \frac{\boldsymbol{x}}{\|\boldsymbol{x}\|_2} - \frac{\boldsymbol{x}'}{\|\boldsymbol{x}'\|_2} \right\|_2 \le 2 \frac{\|\boldsymbol{x} - \boldsymbol{x}'\|_2}{\max\{\|\boldsymbol{x}\|_2, \|\boldsymbol{x}'\|_2\}},$$

whence

$$\|\boldsymbol{F}(\boldsymbol{x}) - \boldsymbol{F}(\boldsymbol{x}')\|_2 \le (2L + M)\|\boldsymbol{x} - \boldsymbol{x}'\|_2,$$

as was to be shown. $\qquad\square$

## C  SKELETON ANALYSIS AND CERTIFICATE CONSTRUCTION

In this section, we construct a certificate $g$ for the certificate problem (B.1) in the context of a simple model geometry. We also collect technical estimates relevant to the analysis of the skeleton $\psi$. We point to Appendix A.5.2 for a summary of the operator and function definitions relevant to the certificate problem that we will use below. We will use the notation

$$\hat{\boldsymbol{\Theta}}[g](\boldsymbol{x}) = \int_{\mathcal{M}} \psi \circ \angle(\boldsymbol{x}, \boldsymbol{x}') g(\boldsymbol{x}') \, \mathrm{d}\mu^\infty(\boldsymbol{x}')$$

in this section; we call explicit attention to this notation to avoid confusion with the kernel $\hat{\Theta} = \psi_1 \circ \angle$ that we have defined in the main text for convenience of exposition.

### C.1  CERTIFICATE CONSTRUCTION

To construct a certificate, it suffices to solve the integral equation

$$\hat{\zeta} = \hat{\boldsymbol{\Theta}}[g] \tag{C.1}$$

for a function $g \in L^2_{\mu^\infty}(\mathcal{M})$ and obtain estimates on the norm of $g$. It is useful to consider separately the contributions of integration over the class manifolds $\mathcal{M}_\pm$ in the action of the operator $\hat{\boldsymbol{\Theta}}$: we can write for any $g$

$$\hat{\boldsymbol{\Theta}}[g](\boldsymbol{x}) = \int_{\mathcal{M}_+} \psi \circ \angle(\boldsymbol{x}, \boldsymbol{x}') g(\boldsymbol{x}') \, \mathrm{d}\mu^\infty_+(\boldsymbol{x}') + \int_{\mathcal{M}_-} \psi \circ \angle(\boldsymbol{x}, \boldsymbol{x}') g(\boldsymbol{x}') \, \mathrm{d}\mu^\infty_-(\boldsymbol{x}'),$$

and it then makes sense to further subdivide based on whether the evaluation point $\boldsymbol{x}$ lies in $\mathcal{M}_+$ or $\mathcal{M}_-$, and to introduce the density $\rho$ explicitly by a change of variables. With a slight abuse of notation, we will write $\mathrm{d}\boldsymbol{x}'$ to denote the Riemannian measure on $\mathcal{M}_+$ and $\mathcal{M}_-$, for concision. Because the kernel $\psi \circ \angle$ is symmetric, if we define an operator $\hat{\boldsymbol{\Theta}}_+ : L^2(\mathcal{M}_+) \to L^2(\mathcal{M}_+)$ by

$$\hat{\boldsymbol{\Theta}}_+[g_+](\boldsymbol{x}) = \int_{\mathcal{M}_+} \psi \circ \angle(\boldsymbol{x}, \boldsymbol{x}') g_+(\boldsymbol{x}') \, \mathrm{d}\boldsymbol{x}',$$

an operator $\hat{\boldsymbol{\Theta}}_- : L^2(\mathcal{M}_-) \to L^2(\mathcal{M}_-)$ by

$$\hat{\boldsymbol{\Theta}}_-[g_-](\boldsymbol{x}) = \int_{\mathcal{M}_-} \psi \circ \angle(\boldsymbol{x}, \boldsymbol{x}') g_-(\boldsymbol{x}') \, \mathrm{d}\boldsymbol{x}',$$

and an operator $\hat{\boldsymbol{\Theta}}_\pm : L^2(\mathcal{M}_+) \to L^2(\mathcal{M}_-)$ by

$$\hat{\boldsymbol{\Theta}}_\pm[g_+](\boldsymbol{x}) = \int_{\mathcal{M}_+} \psi \circ \angle(\boldsymbol{x}, \boldsymbol{x}') g_+(\boldsymbol{x}') \, \mathrm{d}\boldsymbol{x}',$$

then we can write the certificate system (C.1) equivalently as the $2 \times 2$ block operator equation

$$\begin{bmatrix} \hat{\zeta}_+ \\ \hat{\zeta}_- \end{bmatrix} = \begin{bmatrix} \hat{\boldsymbol{\Theta}}_+ & \hat{\boldsymbol{\Theta}}_\pm^* \\ \hat{\boldsymbol{\Theta}}_\pm & \hat{\boldsymbol{\Theta}}_- \end{bmatrix} \begin{bmatrix} \rho_+ g_+ \\ \rho_- g_- \end{bmatrix},$$

where we write $\rho_+$ and $\rho_-$ for the restriction of the density $\rho$ to $\mathcal{M}_+$ and $\mathcal{M}_-$, respectively, and where the adjoint operation is viewed as occurring with operators between $L^2(\mathcal{M}_+)$ and $L^2(\mathcal{M}_-)$ (both Hilbert spaces). We will make use of this notation in the sequel.

### C.1.1 TWO CIRCLES

The two circles geometry is a highly-symmetric geometry where $\mathcal{M}_+$ and $\mathcal{M}_-$ are coaxial circles in the upper and lower hemispheres of $\mathbb{S}^2$, each of radius $0 < r < 1$. Here we note that since the skeleton $\psi$ depends only on the angle between points of $\mathbb{S}^{n_0-1}$, the particular embedding of this geometry into $\mathbb{S}^{n_0-1}$ is irrelevant, and it is without loss of generality to consider the geometry in $\mathbb{S}^2$ once we have restricted ourselves to this configuration. We have unit-speed charts, for $t \in [0, 2\pi r]$

$$\boldsymbol{\gamma}_+(t) = \begin{bmatrix} r\cos t/r \\ r\sin t/r \\ \sqrt{1-r^2} \end{bmatrix}, \qquad \boldsymbol{\gamma}_-(t) = \begin{bmatrix} r\cos t/r \\ r\sin t/r \\ -\sqrt{1-r^2} \end{bmatrix},$$

which implies specific forms of the spherical distances

$$\angle\left(\boldsymbol{\gamma}_+(t), \boldsymbol{\gamma}_+(t')\right) = \cos^{-1}\left(r^2\cos\left|\frac{t-t'}{r}\right| + (1-r^2)\right) \tag{C.2}$$

and

$$\angle\left(\boldsymbol{\gamma}_+(t), \boldsymbol{\gamma}_-(t')\right) = \cos^{-1}\left(r^2\cos\left|\frac{t-t'}{r}\right| - (1-r^2)\right), \tag{C.3}$$

with the analogous results for the remaining possible combinations of domains, by symmetry. Because $\hat{\zeta}$ is piecewise constant on each connected component of $\mathcal{M}$, there are constants $C_+, C_-$ such that $C_+ = \hat{\zeta}$ on $\mathcal{M}_+$ and $C_- = \hat{\zeta}$ on $\mathcal{M}_-$. The block-structured system we are interested in solving is then

$$\begin{bmatrix} C_+ \\ C_- \end{bmatrix} = \begin{bmatrix} \hat{\boldsymbol{\Theta}}_+ & \hat{\boldsymbol{\Theta}}_\pm^* \\ \hat{\boldsymbol{\Theta}}_\pm & \hat{\boldsymbol{\Theta}}_- \end{bmatrix} \begin{bmatrix} \rho_+ g_+ \\ \rho_- g_- \end{bmatrix}, \tag{C.4}$$

where subscripts are used to denote the domain of each component of the certificate. The coordinate representations (C.2) and (C.3) show that each of the operators appearing in the $2 \times 2$ matrix in (C.4) is invariant on the circle; we can obtain some useful simplifications by identifying these operators with their coordinate representations. Defining

$$f_r(t) = \cos^{-1}\left(r^2\cos t + (1-r^2)\right),$$
$$g_r(t) = \cos^{-1}\left(r^2\cos t - (1-r^2)\right),$$

and (self-adjoint) operators on $2\pi$-periodic functions $g$ by

$$\mathcal{A}[g](t) = \int_0^{2\pi} \psi \circ f_r(t-t')g(t')\,\mathrm{d}t',$$

$$\mathcal{X}[g](t) = \int_0^{2\pi} \psi \circ g_r(t-t')g(t')\,\mathrm{d}t',$$

by a change of coordinates, it is equivalent to solve the system

$$\begin{bmatrix} r^{-1}C_+ \\ r^{-1}C_- \end{bmatrix} = \begin{bmatrix} \mathcal{A} & \mathcal{X} \\ \mathcal{X} & \mathcal{A} \end{bmatrix} \begin{bmatrix} \rho_+ g_+ \\ \rho_- g_- \end{bmatrix}, \tag{C.5}$$

where we have identified $\rho_+$ and $\rho_-$ with their coordinate representations, and with an abuse of notation used the same notation for the certificate as in (C.4). We can use symmetry properties to determine

$$g_r(t) = \pi - f_r(t - \pi),$$

so for purposes of analysis we need only consider $f_r$. Each of the invariant operators in (C.5) diagonalizes in the Fourier basis, and because the target $\hat{\zeta}$ is a piecewise constant function, we only need to use the first Fourier coefficient. In other words, we can solve this system by first inverting the invariant operator, which responds to only the constant component of the target, and then inverting the density multiplication operators. This approach is made precise in the following lemma.

**Lemma C.1.** *There is an absolute constant $K > 0$ such that if $L \geq \max\{K, (\pi/2)(1-r^2)^{-1/2}\}$ and $r \geq \frac{1}{2}$, then the system (C.4) has a solution that satisfies*

$$\left\| \begin{bmatrix} g_+ \\ g_- \end{bmatrix} \right\|_{L^2_\mu \infty} \leq \frac{64\left\|\hat{\zeta}\right\|_{L^\infty(\mathcal{M})}}{n\pi^{1/2}\rho_{\min}^{1/2}}.$$

*Proof.* Following the discussion by (C.5), it is equivalent to solve the system in the Fourier basis, with only the DC component. We thus start by solving the system

$$
\begin{bmatrix} r^{-1}C_+ \\ r^{-1}C_- \end{bmatrix} = \begin{bmatrix} 2\int_0^\pi \psi \circ f_r(t)\,\mathrm{d}t & 2\int_0^\pi \psi \circ g_r(t)\,\mathrm{d}t \\ 2\int_0^\pi \psi \circ g_r(t)\,\mathrm{d}t & 2\int_0^\pi \psi \circ f_r(t)\,\mathrm{d}t \end{bmatrix} \begin{bmatrix} G_+ \\ G_- \end{bmatrix},
$$

where $G_+$ and $G_-$ are constants that we will show exist. This is a $2 \times 2$ system, and the matrix is symmetric, with minimum eigenvalue $2\int_0^\pi (\psi \circ f_r - \psi \circ g_r)(t)\,\mathrm{d}t$. Using Lemma C.2, we have if $L \geq \max\{K, (\pi/2)(1-r^2)^{-1/2}\}$ and $r \geq \frac{1}{2}$

$$
2\int_0^\pi (\psi \circ f_r - \psi \circ g_r)(t)\,\mathrm{d}t \geq \frac{\pi n}{32r},
$$

so the $2 \times 2$ matrix is invertible, and by an operator norm bound on its inverse we have the regularity estimate

$$
\left(G_+^2 + G_-^2\right)^{1/2} \leq \frac{32}{\pi n}\left(C_+^2 + C_-^2\right)^{1/2}.
$$

It follows that the function

$$
\begin{bmatrix} g_+ \\ g_- \end{bmatrix} = \begin{bmatrix} \frac{G_+}{\rho_+} \\ \frac{G_-}{\rho_-} \end{bmatrix}
$$

solves the system (C.4). We conclude

$$
\left\| \begin{bmatrix} g_+ \\ g_- \end{bmatrix} \right\|_{L^2_{\mu\infty}}^2 = \int_0^{2\pi} \left( \frac{G_+}{\rho_+ \circ \gamma_+(t)} \right)^2 \rho_+ \circ \gamma_+(t)\,\mathrm{d}t + \int_0^{2\pi} \left( \frac{G_-}{\rho_- \circ \gamma_-(t)} \right)^2 \rho_- \circ \gamma_-(t)\,\mathrm{d}t
$$

$$
\leq \frac{2^{11}}{\pi n^2 \rho_{\min}} \left( C_+^2 + C_-^2 \right).
$$

Taking square roots on both sides of the expression resulting from the last inequality will give the claim, after we simplify the expression $\sqrt{C_+^2 + C_-^2}$. Since

$$
\sqrt{C_+^2 + C_-^2} \leq \sqrt{2}\max\{C_+, C_-\} = \sqrt{2}\left\| \hat{\varsigma} \right\|_{L^\infty(\mathcal{M})},
$$

we can conclude after adjusting constants. $\qquad\square$

**Lemma C.2.** *There exists an absolute constant $K > 0$ such that if $L \geq \max\{K, (\pi/2)(1-r^2)^{-1/2}\}$ and if $r \geq \frac{1}{2}$, one has*

$$
2\int_{[0,\pi]} (\psi \circ f_r - \psi \circ g_r)(t)\,\mathrm{d}t \geq \frac{\pi n}{32r}.
$$

*Proof.* Write $\sigma_r = \psi \circ f_r - \psi \circ g_r$ for brevity, which is nonnegative, by Lemma C.3. We consider the tangent line to the graph of $\sigma_r$ at 0; by Lemma C.3, this line has the form $t \mapsto \sigma_r(0) - tnrL(L+1)/4\pi$, and its graph hits the horizontal axis at $t = 4\pi\sigma_r(0)/nrL(L+1)$. Using that $\sigma_r(0) \leq \psi(0) = nL/2$, we see that this point of intersection is no larger than $2\pi/r(L+1)$, which can be made less than $K$ by choosing $L \geq K'$, where $K > 0$ is the absolute constant appearing in the convexity bound of Lemma C.3, and $K' > 0$ is an absolute constant. Under this condition, we obtain using Lemma C.3

$$
\sigma_r(t) \geq \sigma_r(0) - tnrL(L+1)/4\pi,
$$

and so

$$
\int_{[0,\pi]} \sigma_r(t)\,\mathrm{d}t \geq \int_{[0,4\pi\sigma_r(0)/nrL(L+1)]} (\sigma_r(0) - ntrL(L+1)/4\pi)\,\mathrm{d}t
$$

$$
= \frac{2\pi\sigma_r(0)^2}{nrL(L+1)}.
$$

We have $\sigma_r(0) = nL/2 - \psi(\cos^{-1}(2r^2 - 1))$, and using the estimate of Lemma C.20, we get

$$
\psi(\nu) \leq \frac{nL}{2}\frac{1 + L\nu/2\pi}{1 + L\nu/\pi}.
$$

Together with the estimate $\cos^{-1}(2r^2 - 1) \geq 2\sqrt{1 - r^2}$, we obtain

$$\sigma_r(0) = \frac{nL}{2} - \psi(\cos^{-1}(2r^2 - 1)) \geq \frac{nL}{2}\left(\frac{L\sqrt{1 - r^2}}{\pi + 2L\sqrt{1 - r^2}}\right) \geq \frac{nL}{8},$$

where the final inequality requires the choice $L \geq \pi/2\sqrt{1 - r^2}$. Thus, we have shown

$$\int_{[0,\pi]} \sigma_r(t)\,dt \geq \frac{\pi n}{64r},$$

as claimed. □

**Lemma C.3.** *There is an absolute constant $0 < K \leq \pi/2$ such that if $L \geq 3$, one has for all $r \in (0, 1)$:*

    *(i) $\psi \circ f_r - \psi \circ g_r \geq 0$ on $[0, \pi]$;*

    *(ii) $(\psi \circ f_r - \psi \circ g_r)'(0) = -nrL(L + 1)/4\pi$;*

    *(iii) $\psi \circ f_r - \psi \circ g_r$ is convex on $[0, K]$.*

*Proof.* In this proof, we will make use of basic results on the skeleton $\psi$, namely Lemmas E.5, C.17 and C.18 without making explicit reference to them. Property (i) follows from the fact that $\psi$ is decreasing, $\cos^{-1}$ is decreasing, and the definitions of $f_r$ and $g_r$. We note that $f_r$ is smooth on $(0, \pi)$; to prove property (ii), it will suffice to show that $f_r$ admits a right derivative at $0$ and $\pi$ and apply the chain rule. We have if $t \in (0, \pi)$

$$f_r'(t) = \frac{r^2 \sin t}{\sqrt{1 - (r^2 \cos t + (1 - r^2))^2}} = \frac{1}{\sqrt{2 + r^2(\cos t - 1)}} \frac{r \sin t}{\sqrt{1 - \cos t}}$$

after some rearranging, and by periodicity and symmetry properties of $f_r$, we have

$$\lim_{t \searrow 0} g_r'(t) = \lim_{t \nearrow \pi} f_r'(t) = 0.$$

We Taylor expand $\sin t(1 - \cos t)^{-1/2}$ in a neighborhood of zero to evaluate the derivatives there. We have $\sin t = t - t^3/6 + O(t^5)$ and $1 - \cos t = t^2/2(1 - t^2/2 + O(t^4))$; by the binomial series, we have $(1 - \cos t)^{-1/2} = \sqrt{2}/t(1 + t^2/4 + O(t^4))$, whence $\sin t(1 - \cos t)^{-1/2} = \sqrt{2} + \sqrt{2}t^2/12 + O(t^4)$, and

$$\lim_{t \searrow 0} f_r'(t) = r.$$

Thus

$$(\psi \circ f_r - \psi \circ g_r)'(0) = \psi'(0)f_r'(0) = -\frac{nrL(L + 1)}{4\pi}.$$

For property (iii), now consider $t \in [0, \pi/2]$ when necessary. The chain rule gives

$$(\psi \circ f_r - \psi \circ g_r)'' = [(\psi' \circ f_r)f_r'' - (\psi' \circ g_r)g_r''] + [(\psi'' \circ f_r)(f_r')^2 - (\psi'' \circ g_r)(g_r')^2],$$

and we have if $t \in (0, \pi)$

$$f_r''(t) = \frac{r^4(1 - \cos t)\left[\cos t - (r^2 \cos t + (1 - r^2))\right]}{\left(1 - (r^2 \cos t + (1 - r^2))^2\right)^{3/2}}$$

after some rearranging of the numerator. We have $1 - \cos t \geq 0$, and so the estimate $r^2 \cos t + (1 - r^2) \geq \cos t$ (with equality only at $t = 0$) yields $f_r''(t) \leq 0$ (with a strict inequality if $0 < t < \pi$). By symmetry, this implies that $g_r'' \geq 0$, and using that $\psi' \leq 0$, we obtain

$$(\psi \circ f_r - \psi \circ g_r)'' \geq (\psi'' \circ f_r)(f_r')^2 - (\psi'' \circ g_r)(g_r')^2.$$

By symmetry, we have $g_r(t) = f_r(\pi - t)$ on $[0, \pi]$, and because $f_r$ is strictly concave we know as well that $f_r'$ is strictly decreasing; it follows that $f_r' - g_r'$ is also strictly decreasing, and its unique zero satisfies the equation

$$\frac{1 - \cos t}{1 + \cos t} = \frac{2 - r^2(1 + \cos t)}{2 - r^2(1 - \cos t)}.$$

Noting that $t = \pi/2$ satisfies this equation, we conclude that $f'_r \geq g'_r$ on $[0, \pi/2]$, so that on this interval we have

$$(\psi \circ f_r - \psi \circ g_r)'' \geq (g'_r)^2 ((\psi'' \circ f_r) - (\psi'' \circ g_r)).$$

By Lemma C.19, if $L \geq 3$ there is an absolute constant $K > 0$ such that $\dddot{\psi} \leq 0$ on $[0, K]$. The previous bound then yields

$$(\psi \circ f_r - \psi \circ g_r)'' \geq 0,$$

as claimed. □

## C.2  AUXILIARY RESULTS

### C.2.1  GEOMETRIC RESULTS

**Lemma C.4.** *Let $\mathcal{M}$ be a complete Riemannian submanifold of the unit sphere $\mathbb{S}^{n_0-1}$ (with respect to the spherical metric induced by the euclidean metric on $\mathbb{R}^{n_0}$) with finitely many connected components $K$. If $d_0 = 1$, assume moreover that each connected component of $\mathcal{M}$ is a smooth regular curve. Then for every $0 < \varepsilon \leq 1$, there is a $\varepsilon$-net for $\mathcal{M}$ in the euclidean metric $\|\cdot\|_2$ having cardinality no larger than $(C_{\mathcal{M}}/\varepsilon)^{d_0}$, where $C_{\mathcal{M}} \geq 1$ is a constant depending only on the diameters $\sup_{\boldsymbol{x},\boldsymbol{x}' \in \mathcal{M}_i} \mathrm{dist}_{\mathcal{M}_i}(\boldsymbol{x}, \boldsymbol{x}')$ and, when $d_0 \geq 2$, additionally on the extremal Ricci curvatures of $\mathcal{M}_i$. Moreover, these nets have the property that if $\boldsymbol{x} \in \mathcal{M}$ is given, there is a point in the net $\bar{\boldsymbol{x}}$ within euclidean distance $\varepsilon$ of $\boldsymbol{x}$ such that $\bar{\boldsymbol{x}}$ lies in the same connected component of $\mathcal{M}$ as $\boldsymbol{x}$.*

*Proof.* Consider a fixed connected component $\mathcal{M}_i$ with $i \in [K]$. We write the Riemannian distance of $\mathcal{M}_i$ as $\mathrm{dist}_{\mathcal{M}_i}$; because $\mathcal{M}_i$ is a Riemannian submanifold of $\mathbb{R}^{n_0}$, we have $\mathrm{dist}_{\mathcal{M}_i}(\boldsymbol{x}, \boldsymbol{y}) \geq \|\boldsymbol{x} - \boldsymbol{y}\|_2$ for every $\boldsymbol{x}, \boldsymbol{y}$ in $\mathcal{M}_i$. Because $\mathrm{dist}_{\mathcal{M}_i}(\boldsymbol{x}, \boldsymbol{y}) \geq \|\boldsymbol{x} - \boldsymbol{y}\|_2$, it suffices to estimate the covering number in terms of the Riemannian distance. We will consider distinctly the cases $d_0 = 1$ and $d_0 \geq 2$, starting with $d_0 = 1$.

When $d_0 = 1$, we have assumed that $\mathcal{M}_i$ are regular curves, so it is without loss of generality to assume they are moreover unit-speed curves parameterized by arc length, with lengths $\mathrm{len}(\mathcal{M}_i)$. It follows that we can obtain an $\varepsilon$-net for $\mathcal{M}_i$ in terms of $\mathrm{dist}_{\mathcal{M}_i}$ having cardinality at most $\mathrm{len}(\mathcal{M}_i)/\varepsilon$ when $0 < \varepsilon \leq 1$, and by the submanifold property these sets also constitute $\varepsilon$-nets for $\mathcal{M}_i$ in terms of the $\ell^2$ distance. Covering each connected component $\mathcal{M}_i$ in this way gives a $\varepsilon$-net for $\mathcal{M}$ by taking the union of each connected component's net.

When $d_0 \geq 2$, we make use of standard results relating the covering number to the curvature and diameter of $\mathcal{M}$. Let $\mathrm{diam}(\mathcal{M}_i) = \sup_{\boldsymbol{x},\boldsymbol{x}' \in \mathcal{M}_i} \mathrm{dist}_{\mathcal{M}_i}(\boldsymbol{x}, \boldsymbol{x}')$, and let $\mathrm{Ric}_i$ denote the Ricci curvature tensor of $\mathcal{M}_i$ (recall that we assume the metric on $\mathcal{M}$ is the one induced by the euclidean metric). Then because $\mathcal{M}$ is compact, (1) $\max_{i \in [K]} \mathrm{diam}(\mathcal{M}_i) < +\infty$; and (2) because $\mathrm{Ric}_i$ is moreover continuous, there are constants $k_i > 0$ such that $\mathrm{Ric}_i \geq -(d_0 - 1)k_i$ for each $i \in [K]$. Applying Lemma C.5, it follows that for any $\varepsilon > 0$, there is a $\varepsilon$-net for $\mathcal{M}_i$ in terms of $\mathrm{dist}_{\mathcal{M}_i}$ with cardinality no larger than $(C_{\mathcal{M}_i}/\varepsilon)^{d_0}$, where $C_{\mathcal{M}_i} \lesssim \mathrm{diam}(\mathcal{M}_i)e^{2\,\mathrm{diam}(\mathcal{M}_i)\sqrt{k_i}}$.

Thus, for any $i \in [K]$, any $d_0 \geq 1$ and any $0 < \varepsilon \leq 1$, we can conclude that there is a $\varepsilon$-net for $\mathcal{M}_i$ in the euclidean metric having cardinality no larger than $(C_{\mathcal{M}_i}/\varepsilon)^{d_0}$, where

$$C_{\mathcal{M}_i} = \begin{cases} \mathrm{len}(\mathcal{M}_i) & d_0 = 1 \\ 16\,\mathrm{diam}(\mathcal{M}_i)e^{2\,\mathrm{diam}(\mathcal{M}_i)\sqrt{k_i}} & d_0 \geq 2. \end{cases}$$

Taking the union of these nets and applying Lemma G.10 for simplicity, we conclude that for any $d_0 \geq 1$ and any $0 < \varepsilon \leq 1$, there is a $\varepsilon$-net for $\mathcal{M}$ in the euclidean metric having cardinality no larger than $(C_{\mathcal{M}}/\varepsilon)^{d_0}$, where

$$C_{\mathcal{M}} = \begin{cases} 1 + \sum_{i=1}^{K} \mathrm{len}(\mathcal{M}_i) & d_0 = 1 \\ 1 + 16\sum_{i=1}^{K} \mathrm{diam}(\mathcal{M}_i)e^{2\,\mathrm{diam}(\mathcal{M}_i)\sqrt{k_i}} & d_0 \geq 2. \end{cases}$$

The additional property claimed is satisfied by our construction of the nets. □

**Lemma C.5.** *Given $k > 0$ and integer $d \geq 2$, suppose that $\mathcal{M}$ is a $d$-dimensional complete Riemannian manifold with Ricci curvature tensor satisfying $\mathrm{Ric} \geq -(d-1)k$. Then for any $r, \varepsilon > 0$*

*and any $\boldsymbol{p} \in \mathcal{M}$, there exists an $\varepsilon$-net (measured in the Riemannian distance $\operatorname{dist}_{\mathcal{M}}$) of the metric ball $\{\boldsymbol{x} \in \mathcal{M} \mid \operatorname{dist}_{\mathcal{M}}(\boldsymbol{p}, \boldsymbol{x}) \leq r\}$ with cardinality at most $(C_{\mathcal{M}}/\varepsilon)^d$, where $C_{\mathcal{M}} > 0$ is a constant depending only on $k$ and $r$.*

*Proof.* The proof is essentially an application of (Zhu, 1997, Lemma 3.6) together with some calculations on volumes of geodesic balls in hyperbolic space that we record here for completeness, although they are classical. For any $r > 0$ and any $\boldsymbol{p} \in \mathcal{M}$, write

$$B_r(\boldsymbol{p}) = \{\boldsymbol{x} \in \mathcal{M} \mid \operatorname{dist}_{\mathcal{M}}(\boldsymbol{p}, \boldsymbol{x}) \leq r\}.$$

Fix $\boldsymbol{p} \in \mathcal{M}$ and $r, \varepsilon > 0$. The hypotheses of the lemma make (Zhu, 1997, Lemma 3.6) applicable, whence

$$\inf\left\{\operatorname{card}(S) \,\middle|\, S \subset B_r(\boldsymbol{p}), B_r(\boldsymbol{p}) \subset \bigcup_{\boldsymbol{p}' \in S} B_\varepsilon(\boldsymbol{p}')\right\} \leq \frac{\operatorname{vol}(B^k(2r))}{\operatorname{vol}(B^k(\varepsilon/4))},$$

where $\operatorname{card}(S)$ denotes the cardinality of a set $S$, and for all $\varepsilon > 0$, $\operatorname{vol}(B^k(\varepsilon))$ denotes the volume of a geodesic ball of radius $r$ in the $d$-dimensional simply-connected hyperbolic space of constant sectional curvature $-k$; these spaces are homogeneous and isotropic so the base point does not matter (c.f. (Lee, 2018, Proposition 3.9)). In particular, we can calculate these volumes in any model of hyperbolic space and anchored at any base point; we choose the Poincaré ball model and the base point $\boldsymbol{0}$, where the maximal unit-speed geodesics take the simple form

$$\boldsymbol{\gamma}(t) = k^{-1/2} \boldsymbol{v} \tanh \frac{\sqrt{k}t}{2}$$

for $\boldsymbol{v} \in \mathbb{S}^d$ and $t \in \mathbb{R}$ (Lee, 2018, Theorem 3.7, Proposition 5.28). Integrating the volume form in coordinates, we then get for any $\varepsilon > 0$

$$\operatorname{vol}(B^k(\varepsilon)) = \int_{(k^{-1/2} \tanh \sqrt{k}\varepsilon/2)\mathbb{B}^d} \left(\frac{2/k}{1/k - \|\boldsymbol{x}\|_2^2}\right)^d \mathrm{d}\boldsymbol{x}$$

$$= k^{-d/2} \int_{(\tanh \sqrt{k}\varepsilon/2)\mathbb{B}^d} \left(\frac{2}{1 - \|\boldsymbol{x}\|_2^2}\right)^d \mathrm{d}\boldsymbol{x}$$

where the second line changes coordinates $\boldsymbol{x} \mapsto k^{-1/2}\boldsymbol{x}$. Changing to polar coordinates in the last expression, we get

$$\operatorname{vol}(B^k(\varepsilon)) = k^{-d/2} \operatorname{vol}(\mathbb{S}^{d-1}) \int_{[0, \tanh \sqrt{k}\varepsilon/2]} x^{d-1} \left(\frac{2}{1 - x^2}\right)^d \mathrm{d}x,$$

and then changing coordinates $x \mapsto \tanh x$, we obtain after applying several trigonometric identities

$$\operatorname{vol}(B^k(\varepsilon)) = k^{-d/2} \operatorname{vol}(\mathbb{S}^{d-1}) \int_{[0, \sqrt{k}\varepsilon/2]} 2 \sinh^{d-1}(2x) \, \mathrm{d}x$$

$$= k^{-d/2} \operatorname{vol}(\mathbb{S}^{d-1}) \int_{[0, \sqrt{k}\varepsilon]} \sinh^{d-1}(x) \, \mathrm{d}x,$$

whence

$$\frac{\operatorname{vol}(B^k(2r))}{\operatorname{vol}(B^k(\varepsilon/4))} = \frac{\int_{[0, 2r\sqrt{k}]} \sinh^{d-1}(x) \, \mathrm{d}x}{\int_{[0, \varepsilon\sqrt{k}/4]} \sinh^{d-1}(x) \, \mathrm{d}x}.$$

We have bounds $x \leq \sinh x \leq xe^x$ for nonnegative $x$,[9] which gives after integration

$$\frac{\operatorname{vol}(B^k(2r))}{\operatorname{vol}(B^k(\varepsilon/4))} \leq \frac{\int_{[0, 2r\sqrt{k}]} x^{d-1} e^{(d-1)x} \, \mathrm{d}x}{\int_{[0, \varepsilon\sqrt{k}/4]} x^{d-1} \, \mathrm{d}x}$$

$$\leq d \frac{(2r\sqrt{k})^d \int_{[0,1]} x^{d-1} e^{2r\sqrt{k}(d-1)x} \, \mathrm{d}x}{(\varepsilon\sqrt{k}/4)^d}$$

$$\leq \left(\frac{16re^{2r\sqrt{k}}}{\varepsilon}\right)^d,$$

---

[9]The lower bound is implied by $\cosh x \geq 1$; the upper bound follows from writing $\sinh x = 0.5e^x(1 - e^{-2x})$ and using $e^{-x} \geq 1 - x$.

where in the second line we change coordinates $x \mapsto (2r\sqrt{k})x$, and then use $L^\infty$ control of the (monotone increasing) integrand in the second line to move to the expression in the third line. $\qquad\square$

*Remark* C.6. The constant $C_\mathcal{M}$ in Lemma C.5 can be sharpened if more is known about the curvature of $\mathcal{M}$: if $\mathrm{Ric} \geq 0$, the exponential dependence on curvature and diameter can be removed (intuitively, taking $k \searrow 0$ "recovers" this from the proved result), and if $\mathrm{Ric} > 0$, the dependence on diameter can be completely removed using Myers' theorem (Zhu, 1997, Theorem 3.4(1)).

**Lemma C.7.** *For any $x, x', \bar{x}, \bar{x}'$ in $\mathbb{S}^{n_0-1}$, one has*
$$|\angle(x, x') - \angle(\bar{x}, \bar{x}')| \leq \sqrt{2}\|\|x - x'\|_2 - \|\bar{x} - \bar{x}'\|_2|.$$

*Proof.* Writing $\angle(x, x') = \cos^{-1}\langle x, x'\rangle = \cos^{-1}(1-(1/2)\|x-x'\|_2^2)$, consider the function $f(x) = \cos^{-1}(1 - (1/2)x^2)$ for $x \in [-\sqrt{2}, \sqrt{2}]$, which is differentiable except possibly at 0. We calculate
$$f'(x) = \frac{x}{\sqrt{1 - (1 - \frac{1}{2}x^2)^2}} = \frac{\mathrm{sign}\, x}{\sqrt{1 - \frac{1}{4}x^2}},$$

and taking limits at 0 shows that $f$ admits left and right derivatives on all of $[-\sqrt{2}, \sqrt{2}]$. $f'$ is even-symmetric, so by checking values at 0 and $\sqrt{2}$ we conclude that $|f'| \leq \sqrt{2}$, which shows that $f$ is $\sqrt{2}$-Lipschitz. The claim follows. $\qquad\square$

**Lemma C.8.** *Let $d_0 = 1$. Choose $L$ so that $L \geq K\kappa^2 C_\lambda$, where $\kappa$ and $C_\lambda$ are respectively the curvature and global regularity constants defined in Section 2.1, and $K, K' > 0$ are absolute constants. Then*
$$\sup_{x \in \mathcal{M}_\pm} \int_\mathcal{M} \frac{\mathrm{d}\mu^\infty(x')}{(1 + (L/\pi)\angle(x, x'))^2} \leq \frac{C\rho_{\max}(\mathrm{len}(\mathcal{M}_+) + \mathrm{len}(\mathcal{M}_-))}{L},$$

*where $C$ is an absolute constant and $\mathcal{M}_\pm$ denotes either $\mathcal{M}_+$ or $\mathcal{M}_-$.*

*Proof.* Recall that $\gamma_+, \gamma_-$ denote unit-speed curves parameterized with respect to arc length whose images are $\mathcal{M}_+, \mathcal{M}_-$. For convenience, define $g(\nu) = 1/(1 + L\nu/\pi)$. We have
$$\sup_{x \in \mathcal{M}_\pm} \int_\mathcal{M} (g(\angle(x, x')))^2 \, \mathrm{d}\mu^\infty(x') \leq \begin{array}{l} \sup_{x \in \mathcal{M}_\pm} \int_{\mathcal{M}_+} (g(\angle(x, x')))^2 \, \mathrm{d}\mu_+^\infty(x') \\ + \sup_{x \in \mathcal{M}_\pm} \int_{\mathcal{M}_-} (g(\angle(x, x')))^2 \, \mathrm{d}\mu_-^\infty(x'). \end{array} \qquad (C.6)$$

First, we note that $|g|$ is strictly decreasing. We claim that for any $x \in \mathcal{M}_-$, there is a $x_\star \in \mathcal{M}_+$ such that $\angle(x_\star, x') \leq \angle(x, x')$ for all $x' \in \mathcal{M}_+$; it is easy to see this is the case by choosing $x_\star$ to achieve the minimum in $\min_{x' \in \mathcal{M}_+} \angle(x, x')$ and arguing by contradiction. By monotonicity of the integral, this implies
$$\sup_{x \in \mathcal{M}_\pm} \int_{\mathcal{M}_+} (g(\angle(x, x')))^2 \, \mathrm{d}\mu_+^\infty(x') \leq \sup_{x \in \mathcal{M}_+} \int_{\mathcal{M}_+} (g(\angle(x, x')))^2 \, \mathrm{d}\mu_+^\infty(x'), \qquad (C.7)$$

and similarly for the term involving integration over $\mathcal{M}_-$. Therefore
$$\sup_{x \in \mathcal{M}_\pm} \int_\mathcal{M} (g(\angle(x, x')))^2 \, \mathrm{d}\mu^\infty(x') \leq \begin{array}{l} \sup_{x \in \mathcal{M}_+} \int_{\mathcal{M}_+} (g(\angle(x, x')))^2 \, \mathrm{d}\mu_+^\infty(x') \\ + \sup_{x \in \mathcal{M}_-} \int_{\mathcal{M}_-} (g(\angle(x, x')))^2 \, \mathrm{d}\mu_-^\infty(x'), \end{array} \qquad (C.8)$$

and it suffices to analyze these two terms. We bound the first term, since the second can be bounded by an identical argument. By compactness, the supremum in this term is attained at some $x \in \mathcal{M}_+$. Taking $t$ such that $\gamma_+(t) = x$, we can write
$$\sup_{x \in \mathcal{M}_+} \int_{\mathcal{M}_+} g(\angle(x, x'))^2 \, \mathrm{d}\mu_+^\infty(x') \leq \rho_{\max} \int_0^{S_+} g(\angle(\gamma(t), \gamma(s)))^2 \, \mathrm{d}s. \qquad (C.9)$$

We split the interval $[0, S_+]$ into two disjoint sub-intervals $[0, S_+] \cap [t - K_\tau/\sqrt{L}, t + K_\tau/\sqrt{L}]$ and $[0, S_+] \setminus [t - K_\tau/\sqrt{L}, t + K_\tau/\sqrt{L}]$, corresponding to "large scale" and "small scale" behavior, where $K_\lambda$ is the global regularity constant defined in (A.2). If we now assume $\frac{1}{\sqrt{L}} \leq \frac{c_\lambda}{\kappa}$, then from (A.2) we obtain

$$\angle(\boldsymbol{x}, \boldsymbol{x}') \leq \frac{1}{\sqrt{L}} \Rightarrow \mathrm{dist}_{\mathcal{M}}(\boldsymbol{x}, \boldsymbol{x}') \leq \frac{K_\lambda}{\sqrt{L}}$$

and hence

$$\mathrm{dist}_{\mathcal{M}}(\boldsymbol{x}, \boldsymbol{x}') > \frac{K_\lambda}{\sqrt{L}} \Rightarrow \angle(\boldsymbol{x}, \boldsymbol{x}') > \frac{1}{\sqrt{L}}.$$

From the definition of $g$ it follows that

$$g(\frac{1}{\sqrt{L}}) = \frac{1}{1 + \sqrt{L}/\pi} \leq \frac{\pi}{\sqrt{L}}.$$

Since $|g|$ is a monotonically decreasing function we can bound the second integral in (C.9), obtaining

$$\int_{s \in [0, S_+] \setminus [t^* - \frac{K_\lambda}{\sqrt{L}}, t^* + \frac{K_\lambda}{\sqrt{L}}]} (g(\angle(\boldsymbol{\gamma}(s), \boldsymbol{\gamma}(t^*)))^2 \, ds \leq \mathrm{len}(\mathcal{M}_+) C'/L. \tag{C.10}$$

We next consider the remaining interval of integration in (C.9). Defining

$$S_+^+ = \min\left\{\frac{K_\lambda}{\sqrt{L}}, \quad S_+ - t^*\right\}, \quad S_+^- = \min\left\{\frac{K_\lambda}{\sqrt{L}}, t^*\right\},$$

and $\nu_\pm(s) = \angle(\boldsymbol{\gamma}_+(t^* \pm s), \boldsymbol{\gamma}_+(t^*))$, the integral of interest can be written as

$$\rho_{\max} \int_{s \in [0, S_+] \cap [t^* - \frac{K_\tau}{\sqrt{L}}, t^* + \frac{K_\tau}{\sqrt{L}}]} (g(\angle(\boldsymbol{\gamma}(s), \boldsymbol{\gamma}(t^*))))^2 \, ds = \rho_{\max} \int_{s=0}^{S_+^+} (g(\nu_+(s)))^2 \, ds$$

$$+ \rho_{\max} \int_{s=0}^{S_+^-} (g(\nu_-(s)))^2 \, ds. \tag{C.11}$$

It will be sufficient to consider the first integral here since the second one can be bounded in an identical fashion. We aim to show that the integral above is not too large. This will be the case if $\nu_+(s)$ stays very small for a large range of values of $s$. To show that this is does not occur, we will use our bounds on the curvature of $\mathcal{M}$ to bound $\nu_+(s)$ uniformly from below, which will in turn provide an upper bound on the integral. We will require an application of Lemma C.9, which will be applicable if $S_+^+ \leq \frac{\pi}{\kappa}$. If $L \geq \frac{\kappa^2 K_\tau^2}{\pi^2}$ we have

$$S_+^+ \leq \frac{K_\lambda}{\sqrt{L}} \leq \frac{\pi}{\kappa}.$$

It follows immediately that Lemma C.9 applies to any restriction of $\boldsymbol{\gamma}_+$ of length no larger than $\frac{\pi}{\kappa}$. Next define by $\tilde{\boldsymbol{\gamma}} : [0, S_+^+] \to \mathbb{S}^{n_0 - 1}$ an unit-speed arc of curvature $\kappa$, and $\tilde{\nu}(s) = \angle(\tilde{\boldsymbol{\gamma}}(0), \tilde{\boldsymbol{\gamma}}(s))$.

We claim that

$$\forall s \in [0, S_+^+] : \quad \nu_+(s) \geq \tilde{\nu}(s). \tag{C.12}$$

The proof is by contradiction. Assume there is some $r$ such that

$$\nu_+(r) < \tilde{\nu}(r). \tag{C.13}$$

Now define by $\boldsymbol{\gamma}_r : [0, r] \to \mathbb{S}^{n_0 - 1}$ a restriction of $\boldsymbol{\gamma}_+$ such that $\boldsymbol{\gamma}_r(0) = \boldsymbol{\gamma}_+(t^*), \boldsymbol{\gamma}_r(s) = \boldsymbol{\gamma}_+(t^* + s)$, by $\breve{\boldsymbol{\gamma}}_r$ an arc with curvature $\kappa$ and the same endpoints as $\boldsymbol{\gamma}_r$, and by $\tilde{\boldsymbol{\gamma}}_r$ a restriction of $\tilde{\boldsymbol{\gamma}}$ with

$$\mathrm{len}(\tilde{\boldsymbol{\gamma}}_r) = \mathrm{len}(\boldsymbol{\gamma}_r) = r.$$

Note that $\angle(\tilde{\boldsymbol{\gamma}}_r(0), \tilde{\boldsymbol{\gamma}}_r(s)) = \tilde{\nu}(r)$. However, an application of Lemma C.9 gives

$$\mathrm{len}(\boldsymbol{\gamma}_r) \leq \mathrm{len}(\breve{\boldsymbol{\gamma}}_r) < \mathrm{len}(\tilde{\boldsymbol{\gamma}}_r)$$

where the second inequality is because $\breve{\gamma}_r$ and $\tilde{\gamma}_r$ have identical curvature at every point, and by assumption (C.13) the endpoints of $\tilde{\gamma}_r$ are a greater geodesic (and hence euclidean) distance from each other than the endpoints of $\breve{\gamma}_r$ (which are a distance $\nu_+(r)$ apart). This inequality contradicts the equality above it, and we conclude that no such $r$ exists, and (C.12) holds.

We have that $|g|$ is a monotonically decreasing function, hence we can write for the first integral in (C.11)

$$\int_{s=0}^{S_+^+} \left(g(\nu_+(s))\right)^2 ds \leq \int_{s=0}^{S_+^+} \left(g(\tilde{\nu}(s))\right)^2 ds.$$

We now bound this integral. Since $\tilde{\gamma}$ is an arc with curvature $\kappa$, from the proof of Lemma C.3 we have that $\tilde{\nu}$ is concave, and since $\tilde{\nu}(0) = 0$ we can write

$$\tilde{\nu}(s) \geq \frac{\tilde{\nu}(S_+^+)}{S_+^+} s,$$

and since $|g|$ is monotonically decreasing

$$\int_{s=0}^{S_+^+} \left(g(\tilde{\nu}(s))\right)^2 ds \quad \leq \int_{s=0}^{S_+^+} \left(g\left(\frac{\tilde{\nu}(S_+^+)}{S_+^+}s\right)\right)^2 ds \quad = \frac{S_+^+}{\tilde{\nu}(S_+^+)} \int_{s=0}^{\tilde{\nu}(S_+^+)} \left(g(s)\right)^2 ds$$

$$= \frac{S_+^+}{\tilde{\nu}(S_+^+)} \frac{\tilde{\nu}(S_+^+)}{1 + L\tilde{\nu}(S_+^+)/\pi} \quad \leq \pi \frac{S_+^+}{L\tilde{\nu}(S_+^+)}$$

where we used the definition of $g$. It remains to show that $S_+^+$ and $\tilde{\nu}(S_+^+)$ are close. Since $\tilde{\gamma}$ is an arc with curvature $\kappa$ and length $S_+^+$, if we additionally assume $L \geq K\kappa^2 C_\lambda$ for some $K$ chosen so that $\frac{\kappa\|\tilde{\gamma}(0)-\tilde{\gamma}(S_+^+)\|_2}{2} \leq \frac{\kappa S_+^+}{2} \leq \frac{\kappa C_\lambda}{2\sqrt{L}} \leq \frac{1}{2}$, we obtain

$$\left\|\tilde{\gamma}(0) - \tilde{\gamma}(S_+^+)\right\|_2 \quad \leq S_+^+$$

$$= \frac{2}{\kappa} \sin^{-1}\left(\frac{\kappa\left\|\tilde{\gamma}(0) - \tilde{\gamma}(S_+^+)\right\|_2}{2}\right)$$

$$\leq \left\|\tilde{\gamma}(0) - \tilde{\gamma}(S_+^+)\right\|_2 + \frac{\kappa^2}{4}\left\|\tilde{\gamma}(0) - \tilde{\gamma}(S_+^+)\right\|_2^3$$

$$\left|\left\|\tilde{\gamma}(0) - \tilde{\gamma}(S_+^+)\right\|_2 - S_+^+\right| \leq \frac{\kappa^2}{4}\left\|\tilde{\gamma}(0) - \tilde{\gamma}(S_+^+)\right\|_2^3 \leq \frac{\kappa^2}{4}\left(S_+^+\right)^3 \leq \frac{\kappa^2}{4}\frac{K^2}{L}S_+^+$$

where in the first line we used $\sin^{-1}(x) \leq x + x^3$ for $x$. Since

$$\left\|\tilde{\gamma}(0) - \tilde{\gamma}(S_+^+)\right\|_2 \leq \angle(\tilde{\gamma}(0), \tilde{\gamma}(S_+^+)) = \tilde{\nu}(S_+^+) \leq S_+^+$$

we obtain

$$\left|\tilde{\nu}(S_+^+) - S_+^+\right| \leq \frac{\kappa^2}{4}\frac{K_\lambda^2}{L}S_+^+$$

and hence

$$\frac{S_+^+}{\tilde{\nu}(S_+^+)} \leq \frac{S_+^+}{S_+^+ - \frac{\kappa^2}{4}\frac{K_\lambda^2}{L}S_+^+} = \frac{1}{1 - \frac{\kappa^2}{4}\frac{K_\lambda^2}{L}}.$$

We now choose $L \geq K\kappa^2 K_\tau^2$ for some $K$, so that the above term is smaller than 2. We therefore have

$$\int_{s=0}^{S_i} \left(g(\tilde{\nu}(s))\right)^2 ds \leq C/L$$

for some $C$. We can bound the second integral in (C.11) in an identical fashion. Combining this result with (C.10) and recalling (C.9), we obtain

$$\sup_{\boldsymbol{x}\in\mathcal{M}_+} \int_{\mathcal{M}_+} \left(g(\angle(\boldsymbol{x}, \boldsymbol{x}'))\right)^2 d\mu^\infty(\boldsymbol{x}') \leq C'\rho_{\max}(\text{len}(\mathcal{M}_+) + \text{len}(\mathcal{M}_-))/L$$

for some constant, which completes the proof.

□

**Lemma C.9.** *Given a smooth, simple open curve in $\mathbb{R}^n$ of length $S$ with unit-speed parametrization $\boldsymbol{\gamma} : [0, S] \to \mathbb{R}^n$ such that for some $\kappa > 0$*

   *1. $\|\ddot{\boldsymbol{\gamma}}\|_2 \leq \kappa$*

   *2. $S \leq \frac{\pi}{\kappa}$*

*define by $\check{\boldsymbol{\gamma}}$ an arc of any circle of radius $\frac{1}{\kappa}$ such that $\check{\boldsymbol{\gamma}}(0) = \boldsymbol{\gamma}(0), \check{\boldsymbol{\gamma}}(\check{S}) = \boldsymbol{\gamma}(S), \check{S} \leq \frac{\pi}{\kappa}$[10]. We then have*

$$S \leq \check{S}$$

*Proof.* This result is a generalization of a well known comparison theorem of Schur's to higher dimensions following the proof in (Sullivan, 2008), where we additionally specialize to the case where one of the curves is an arc.

Given a curve $\boldsymbol{\gamma}$ satisfying the conditions of the lemma, we first consider an arc $\tilde{\boldsymbol{\gamma}}$ of a circle of radius $\frac{1}{\kappa}$ and length $S$, with a unit-speed parametrization. At the midpoint of this arc, the tangent vector $\tilde{\boldsymbol{\gamma}}'(\frac{S}{2})$ is parallel to $\tilde{\boldsymbol{\gamma}}(S) - \tilde{\boldsymbol{\gamma}}(0)$, hence

$$\|\tilde{\boldsymbol{\gamma}}(S) - \tilde{\boldsymbol{\gamma}}(0)\|_2 = \left\langle \tilde{\boldsymbol{\gamma}}'(\frac{S}{2}), \tilde{\boldsymbol{\gamma}}(S) - \tilde{\boldsymbol{\gamma}}(0) \right\rangle = \left\langle \tilde{\boldsymbol{\gamma}}'(\frac{S}{2}), \int_0^S \tilde{\boldsymbol{\gamma}}'(t)dt \right\rangle.$$

Similarly, for the curve $\boldsymbol{\gamma}$ we have

$$\|\boldsymbol{\gamma}(S) - \boldsymbol{\gamma}(0)\|_2 \geq \left\langle \boldsymbol{\gamma}'(\frac{S}{2}), \boldsymbol{\gamma}(S) - \boldsymbol{\gamma}(0) \right\rangle = \left\langle \boldsymbol{\gamma}'(\frac{S}{2}), \int_0^S \boldsymbol{\gamma}'(t)dt \right\rangle.$$

Denoting the angle between tangent vectors $\langle \boldsymbol{\gamma}'(a), \boldsymbol{\gamma}'(b) \rangle = \cos\theta(a, b)$, we use the fact that for any smooth curve with unit-speed parametrization $\|\boldsymbol{\gamma}''(t)\|_2 = \left| \frac{d\theta}{ds}(t, s)_{s=t} \right| \doteq |\theta'(t)|$. This gives for any $t \in [0, S/2]$

$$
\begin{aligned}
\left\langle \boldsymbol{\gamma}'(\frac{S}{2}), \boldsymbol{\gamma}'(\frac{S}{2} + t) \right\rangle &= \cos\left( \int_{\frac{S}{2}}^{\frac{S}{2}+t} \theta'(t')dt' \right) \\
&\geq \cos\left( \int_{\frac{S}{2}}^{\frac{S}{2}+t} |\theta'(t')| \, dt' \right) = \cos\left( \int_{\frac{S}{2}}^{\frac{S}{2}+t} \|\boldsymbol{\gamma}''(t)\|_2 \, dt' \right) \\
&\geq \cos\left( \kappa \int_{\frac{S}{2}}^{\frac{S}{2}+t} dt' \right) = \cos\left( \int_{\frac{S}{2}}^{\frac{S}{2}+t} \|\tilde{\boldsymbol{\gamma}}''(t)\|_2 \, dt' \right) = \left\langle \tilde{\boldsymbol{\gamma}}'(\frac{S}{2}), \tilde{\boldsymbol{\gamma}}'(\frac{S}{2} + t) \right\rangle
\end{aligned}
$$

where we have used monotonicity of $\cos$ over the relevant range which is ensured by assumption 2, and a similar argument follows for $t \in (0, -S/2]$. Combining these inequalities gives

$$\|\boldsymbol{\gamma}(S) - \boldsymbol{\gamma}(0)\|_2 \geq \|\tilde{\boldsymbol{\gamma}}(S) - \tilde{\boldsymbol{\gamma}}(0)\|_2.$$

We have shown that, unsurprisingly, if the curvature of $\boldsymbol{\gamma}$ is bounded and it is not too long, then the distance between its endpoints is greater than that of a curve of equal length but larger curvature - namely the arc $\tilde{\boldsymbol{\gamma}}$. We now consider the arc $\check{\boldsymbol{\gamma}}$ defined in the lemma statement. If $S > \check{S}$, due to assumption 2 this would imply

$$\|\tilde{\boldsymbol{\gamma}}(S) - \tilde{\boldsymbol{\gamma}}(0)\|_2 > \left\| \check{\boldsymbol{\gamma}}(\check{S}) - \check{\boldsymbol{\gamma}}(0) \right\|_2 = \|\boldsymbol{\gamma}(S) - \boldsymbol{\gamma}(0)\|_2$$

contradicting the inequality proved above. It follows that $S \leq \check{S}$. □

---

[10]For any circle and choice of endpoints there will be two such arcs, and the last condition implies that we choose the shorter of the two.

### C.2.2 ANALYSIS OF THE SKELETON

**Notation.** Define $\varphi^{(0)} = \mathrm{Id}$, and for $\ell \in \mathbb{N}$ define $\varphi^{(\ell)}$ as the $\ell$-fold composition of $\varphi$ with itself, where

$$\varphi(\nu) = \cos^{-1}\left( \left(1 - \pi^{-1}\nu\right) \cos\nu + \pi^{-1} \sin\nu \right)$$

is the heuristic angle evolution function. We will make use of basic properties of this function such as smoothness (established in Lemma E.5) below. In this section, we will study the skeleton

$$\psi_1(\nu) = \frac{n}{2} \sum_{\ell=0}^{L-1} \cos\varphi^{(\ell)}(\nu) \prod_{\ell'=\ell}^{L-1} \left(1 - \pi^{-1}\varphi^{(\ell')}(\nu)\right), \qquad \nu \in [0, \pi],$$

where we have not included the additive factor $\cos\varphi^{(L)}(\nu)$, as it is easily removed along the lines of Theorem B.2. We define

$$\xi^{(\ell)}(\nu) = \prod_{\ell'=\ell}^{L-1} \left(1 - \pi^{-1}\varphi^{(\ell')}(\nu)\right), \qquad \ell = 0, \dots, L-1,$$

so that

$$\psi_1(\nu) = \frac{n}{2} \sum_{\ell=0}^{L-1} \cos\varphi^{(\ell)}(\nu)\xi^{(\ell)}(\nu). \tag{C.14}$$

We will also establish a convenient approximation to the skeleton. Define

$$\psi(\nu) = \frac{n}{2} \sum_{\ell=0}^{L-1} \xi^{(\ell)}(\nu).$$

Lemma C.10 implies that $\psi$ is convex; it is less trivial to obtain the same for $\psi_1$. We will prove several estimates below for the terms $\xi^{(\ell)}$ and their derivatives that can be used to immediately obtain useful estimates for $\psi$ and its derivatives.

**Lemma C.10.** *For each $\ell = 0, 1, \dots, L$, the functions $\varphi^{(\ell)}$ are nonnegative, strictly increasing, and concave (positive and strictly concave on $(0, \pi)$); if $0 \le \ell < L$, the functions $\xi^{(\ell)}$, are nonnegative, strictly decreasing, and convex (positive and strictly convex on $(0, \pi)$).*

*Proof.* These claims are a consequence of some general facts for smooth functions that we articulate here so that we can rely on them often in the sequel. First, we have for any smooth function $f : (0, \pi) \to \mathbb{R}$

$$(f \circ f)' = (f' \circ f)f',$$
$$(f \circ f)'' = (f' \circ f)f'' + (f')^2(f'' \circ f).$$

These equations show that if $f > 0$, $f' > 0$, and $f'' < 0$, then $f \circ f$ also satisfies these three properties. Lemma E.5 shows that $\varphi$ satisfies these three properties on $(0, \pi)$; we conclude from the mean value theorem and a simple induction the same for $\varphi^{(\ell)}$, as claimed. Meanwhile, if $f, g$ are smooth real-valued functions on $(0, \pi)$, we have

$$(fg)' = f'g + g'f,$$
$$(fg)'' = f''g + g''f + 2f'g'.$$

Thus, if $f$ and $g$ are both positive, strictly decreasing, strictly convex functions on $(0, \pi)$, then $fg$ also satisfies these three properties. Lemma E.5 implies that $0 < 1 - \pi^{-1}\varphi^{(\ell)} < 1$ on $(0, \pi)$, and the first and second derivatives are scaled and negated versions of those of $\varphi^{(\ell)}$; we conclude by another induction that the same three properties apply to the functions $\xi^{(\ell)}$. $\qquad \square$

**Lemma C.11.** *There is an absolute constant $C > 0$ such that if $L \ge 12$ and $n \ge L$, then one has*

$$\|\psi_1 - \psi\|_{L^\infty} \le \frac{Cn}{L}.$$

*Proof.* We have from the triangle inequality

$$\|\psi_1 - \psi\|_{L^\infty} \leq \sup_{\nu \in [0,\pi]} \left( \frac{n}{2} \sum_{\ell=0}^{L-1} \left| \cos \varphi^{(\ell)}(\nu) - 1 \right| \xi^{(\ell)}(\nu) \right)$$

$$\leq \frac{n}{2} \sum_{\ell=0}^{L-1} \sup_{\nu \in [0,\pi]} \left( \left| \cos \varphi^{(\ell)}(\nu) - 1 \right| \xi^{(\ell)}(\nu) \right),$$

where we use Lemma C.10 to take $\xi^{(\ell)}$ outside the absolute value. Notice that $(\cos \varphi^{(\ell)} - 1)\xi^{(\ell)} \leq 0$, so to control the $L^\infty$ norm of this term it suffices to bound it from below. We will show the monotonicity property

$$(\cos \varphi^{(\ell)} - 1)\xi^{(\ell)} - (\cos \varphi^{(\ell+1)} - 1)\xi^{(\ell+1)} \geq 0, \tag{C.15}$$

from which it follows

$$\|\psi_1 - \psi\|_{L^\infty} \leq \frac{nL}{2} \sup_{\nu \in [0,\pi]} \left| \cos \varphi^{(L-1)}(\nu) - 1 \right|,$$

using also $\xi^{(L-1)}(\nu) \leq 1$. Since $\cos x \geq 1 - (1/2)x^2$, and since Lemma C.12 gives that $\varphi^{(L-1)} \leq C/(L-1)$ (and also estimates the constant), we have as soon as $L \geq 1 + C/\sqrt{2}$

$$\|\psi_1 - \psi\|_{L^\infty} \leq \frac{C^2 nL}{4(L-1)^2}$$

which gives the claim provided $L \geq 2$ and $n \geq L$. So to conclude, we need only establish (C.15). To this end, write the LHS of (C.15) as

$$(\cos \varphi^{(\ell)} - 1)\xi^{(\ell)} - (\cos \varphi^{(\ell+1)} - 1)\xi^{(\ell+1)} = \left[ (\cos \varphi^{(\ell)} - \cos \varphi^{(\ell+1)}) - \frac{\varphi^{(\ell)}}{\pi}(\cos \varphi^{(\ell)} - 1) \right] \xi^{(\ell+1)}$$

to notice that it suffices to prove nonnegativity of the bracketed quantity. In addition, since $\ell \geq 0$ and $\varphi(\nu) \leq \nu$ by Lemma E.5, we can instead prove the inequality

$$(\cos x - \cos \varphi(x)) - \frac{x}{\pi}(\cos x - 1) \geq 0$$

for all $x \in [0, \pi]$. Using the closed-form expression for $\cos \varphi(x)$ in Lemma E.2, we can plug into the previous inequality and cancel to get the equivalent inequality

$$x - \sin x \geq 0.$$

But this is immediate from the concavity estimate $\sin x \leq x$, and (C.15) is proved. $\qquad \square$

**Lemma C.12.** *If $\ell \in \mathbb{N}_0$, one has the "fluid" estimate for the angle evolution function*

$$\varphi^{(\ell)}(\nu) \leq \frac{\nu}{1 + c\ell\nu},$$

*where $c > 0$ is an absolute constant. In particular, if $\ell \in \mathbb{N}$ one has $\varphi^{(\ell)} \leq 1/c\ell$.*

*Proof.* The second claim follows from the first claim and $1 + c\ell\nu \geq c\ell\nu$, so we will focus on establishing the first estimate. The proof is by induction on $\ell \in \mathbb{N}$, since the case of $\ell = 0$ is immediate. By Lemma E.5, there is a constant $c_1 > 0$ such that $\varphi(\nu) \leq \nu(1 - c_1\nu)$, and using the numerical inequality $x(1-x) \leq x(1+x)^{-1}$, valid for $x \geq 0$, we get

$$\varphi(\nu) \leq \frac{\nu}{1 + c_1\nu}, \tag{C.16}$$

which establishes the claim in the case $\ell = 1$. Assuming the claim holds for $\ell - 1$, we calculate

$$\varphi^{(\ell)}(\nu) \leq \frac{\varphi^{(\ell-1)}(\nu)}{1 + c_1\varphi^{(\ell-1)}(\nu)} \leq \frac{\frac{\nu}{1+c_1(\ell-1)\nu}}{1 + c_1 \frac{\nu}{1+c_1(\ell-1)\nu}},$$

where the first inequality uses (C.16), and the second inequality uses the induction hypothesis and the relation $x(1+x)^{-1} = 1 - (1+x)^{-1}$ to see that $x \mapsto x(1 + c_1x)^{-1}$ is increasing. Clearing denominators in the numerator and denominator of the RHS of this last bound, we see that it is equal to $\nu/(1 + \ell\nu/\pi)$, and the claim follows by induction.

$\qquad \square$

**Lemma C.13.** *If $\ell \in \mathbb{N}_0$, the iterated angle evolution function satisfies the estimate*

$$\varphi^{(\ell)}(\nu) \geq \frac{\nu}{1 + \ell\nu/\pi}.$$

*Proof.* The proof is by induction on $\ell \in \mathbb{N}$, since the case $\ell = 0$ is immediate. The case $\ell = 1$ follows from Lemma C.14. Assuming the claim holds for $\ell - 1$, we calculate

$$\varphi^{(\ell)}(\nu) \geq \frac{\varphi^{(\ell-1)}(\nu)}{1 + \varphi^{(\ell-1)}(\nu)/\pi} \geq \frac{\frac{\nu}{1+(\ell-1)\nu/\pi}}{1 + \frac{1}{\pi}\frac{\nu}{1+(\ell-1)\nu/\pi}},$$

where the first inequality applies Lemma C.14, and the second uses the fact that the RHS of the bound in Lemma C.14 is strictly increasing and the induction hypothesis. Clearing denominators in the numerator and denominator of the RHS of this last bound, we see that it is equal to $\nu/(1+\ell\nu/\pi)$, and the claim follows by induction. $\qquad\square$

**Lemma C.14.** *It holds*

$$\varphi(\nu) \geq \frac{\nu}{1 + \nu/\pi}.$$

*Proof.* After some rearranging using Lemma E.2, it suffices to prove

$$\left(1 - \frac{\nu}{\pi}\right)\cos\nu + \frac{\sin\nu}{\pi} \leq \cos\left(\frac{\pi\nu}{\pi + \nu}\right). \tag{C.17}$$

Using Lemma E.5, we see that both the LHS and RHS of this bound are nonincreasing. We will prove the estimate in three stages, using "small angle", "large angle", and "intermediate angle" estimates of the quantities on both sides of (C.17). Since $\pi\nu/(\pi + \nu) \in [0, \pi/2]$, we can use standard estimates for $\cos$ to get RHS estimates

$$\cos\left(\frac{\pi\nu}{\pi + \nu}\right) \geq 1 - \frac{1}{2}\left(\frac{\pi\nu}{\pi + \nu}\right)^2 \tag{C.18}$$

and

$$\cos\left(\frac{\pi\nu}{\pi + \nu}\right) \geq \frac{\pi - \nu}{\pi + \nu}. \tag{C.19}$$

As for the LHS, we can obtain an estimate near $\nu = \pi$ in a straightforward way. Transforming the domain by $\nu \mapsto \pi - \nu$, it suffices to get estimates on $\sin\nu - \nu\cos\nu$ near $\nu = 0$, then divide by $\pi$. Using $\cos\nu \geq 1 - (1/2)\nu^2$ and $\sin\nu \leq \nu$, it follows that $\sin\nu - \nu\cos\nu \leq (1/2)\nu^3$. We conclude

$$\left(1 - \frac{\nu}{\pi}\right)\cos\nu + \frac{\sin\nu}{\pi} \leq \frac{1}{2\pi}(\pi - \nu)^3. \tag{C.20}$$

We will develop a second-order approximation to the LHS near $0$ for the small-angle estimates. The first, second, and third derivatives of the LHS are $(1 - \nu/\pi)\sin\nu$, $(1/\pi)\sin\nu - (1 - \nu/\pi)\cos\nu$, and $(2/\pi)\cos\nu + (1 - \nu/\pi)\sin\nu$, respectively. To bound the third derivative, we will use the estimate $\cos\nu \leq 1 - \nu^2/3$ on $[0, \pi/2]$. To prove this, note that Taylor's formula implies the bound $\cos\nu \leq 1 - \nu^2/3$ on $[0, \cos^{-1}(2/3)]$; because $\cos$ is concave on $[0, \pi/2]$, we also have the tangent line bound $\cos(\nu) \leq -\nu\sqrt{5}/3 + (2/3 + \sqrt{5}\cos^{-1}(2/3)/3)$ on $[0, \pi/2]$. We can then solve for the zeros of the quadratic polynomial $1 - \nu^2/3 + (\sqrt{5}/3)\nu - (2/3 + \sqrt{5}\cos^{-1}(2/3)/3)$; a numerical evaluation shows that both roots are real and outside the interval $[\cos^{-1}(2/3), \pi/2]$. Since the tangent line touches the graph of $\cos$ at $\nu = \cos^{-1}(2/3)$, this proves that $\cos\nu \leq 1 - \nu^2/3$ on $[0, \pi/2]$. We can therefore write

$$2\cos\nu + (\pi - \nu)\sin\nu \leq 2(1 - \nu^2/3) + \nu(\pi - \nu), \qquad \nu \in [0, \pi/2].$$

The RHS of this inequality is a concave quadratic; we calculate its maximum analytically as $2 + 3\pi^2/20$. Meanwhile, if $\nu \in [\pi/2, \pi]$, we have $2\cos\nu \leq 0$, and $(\pi - \nu)\sin\nu \leq \pi/2$. We conclude that $(2/\pi)\cos\nu + (1 - \nu/\pi)\sin\nu \leq 2 + 3\pi^2/20$ on $[0, \pi]$. Writing $c = 1/(3\pi) + \pi/40$, this implies an estimate

$$\left(1 - \frac{\nu}{\pi}\right)\cos\nu + \frac{\sin\nu}{\pi} \leq 1 - \frac{\nu^2}{2} + c\nu^3. \tag{C.21}$$

Finally, we will need some estimates for interpolating the small and large angle regimes. We note that the second derivative $(1/\pi)\sin\nu - (1 - \nu/\pi)\cos\nu$ of the LHS of (C.17) is nonnegative if $\nu \geq \pi/2$, because $\cos \geq 0$ here; meanwhile, the third derivative $(2/\pi)\cos\nu + (1 - \nu/\pi)\sin\nu$ of the LHS of (C.17) is nonnegative if $0 \leq \nu \leq \pi/2$, since $\cos \geq 0$ here, and it follows that the second derivative is increasing on $[0, \pi/2]$. Checking numerically that the value of the second derivative at $1.42$ is positive, we conclude that the LHS of (C.17) is convex on $[1.42, \pi]$. In addition, we use calculus to evaluate the first and second derivative of the RHS of (C.18) as $-\nu\pi^3/(\pi + \nu)^3$ and $-\pi^3(\pi - 2\nu)/(\pi + \nu)^4$, respectively; this shows that the RHS of (C.18) is convex for $\nu \geq \pi/2$, and concave for $\nu \leq \pi/2$. Taking a tangent line to the graph of the RHS of (C.18) at $\pi/2$, it follows that the function

$$g(x) = \begin{cases} 1 - (\pi^2/2)\nu^2/(\pi + \nu)^2 & x \leq \pi/2 \\ -(4\pi/27)\nu + (1 + \pi^2/54) & x \geq \pi/2 \end{cases} \tag{C.22}$$

is a concave lower bound for the RHS of (C.18) on $[0, \pi]$.

We proceed to using the estimates developed in the previous paragraph to prove (C.17). We first argue that for $\nu$ in a neighborhood of $0$, we have

$$1 - \nu^2/2 + c\nu^3 \leq 1 - (\pi^2/2)\nu^2/(\pi + \nu)^2,$$

which will in turn prove (C.17) in the same neighborhood. Cancelling and rearranging, it is equivalent to show

$$(2/\pi - 2c) - (4c/\pi - 1/\pi^2)\nu - (2c/\pi^2)\nu^2 \geq 0.$$

The LHS is a concave quadratic, with value $2/\pi - 2c > 0$ at $0$; we calculate its two distinct roots numerically as lying in the intervals $[-5.1, -5]$ and $[1.42, 1.43]$, respectively. It follows that (C.17) holds for $\nu \in [0, 1.42]$. Next, we argue that for $\nu$ in a neighborhood of $\pi$, we have

$$\frac{1}{2\pi}(\pi - \nu)^3 \leq \frac{\pi - \nu}{\pi + \nu},$$

which will in turn prove (C.17) in the same neighborhood. Transforming with $\nu \mapsto \pi - \nu$ and rearranging, it is equivalent to show $\nu^2(2\pi - \nu) \leq 4\pi^2$ in a neighborhood of $0$. The LHS of this last inequality is $0$ at $0$, and nonnegative on $[0, \pi]$; its first and second derivatives are $\nu(4\pi - 3\nu)$ and $4\pi - 6\nu$, respectively, which shows that it is a strictly increasing function of $\nu$ on $[0, \pi]$. Verifying numerically the three distinct real roots of $\nu^3 - 2\pi\nu^2 + 1 = 0$ and transferring the result back via another transformation $\nu \mapsto \pi - \nu$, we conclude that (C.17) holds on $[\pi - 1.1, \pi]$. To obtain that (C.17) holds on $[1.42, \pi - 1.1]$, we use that the function $g$ defined in (C.22) is a concave lower bound for the RHS of (C.18), so that it suffices to show that the LHS of (C.17) is upper bounded by $g$ on $[1.42, \pi - 1.1]$. The LHS of (C.17) is convex on $[1.42, \pi]$, so it follows that it is sufficient to show that the values of the LHS of (C.17) at $1.42$ and at $\pi - 1.1$ are upper bounded by those of $g$ at the same points. Confirming this numerically, we can conclude the proof.

$\square$

**Lemma C.15.** *If $\ell \in \mathbb{N}_0$, one has*

$$\left|\dot{\varphi}^{(\ell)}(\nu)\right| \leq \frac{1}{1 + (c/2)\ell\nu},$$

*where $c > 0$ is the absolute constant also appearing in Lemma E.5 (property 4), and in particular $c/2$ is equal to the absolute constant appearing in Lemma C.12. In particular, if $\ell \in \mathbb{N}$ and $\nu \in [0, \pi]$ we have the estimate*

$$\left|\nu\dot{\varphi}^{(\ell)}(\nu)\right| \leq \frac{2}{c\ell}.$$

*Proof.* The case of $\ell = 0$ follows directly (as an equality) from $\varphi^{(0)}(\nu) = \nu$. Now we assume $\ell \in \mathbb{N}$. Smoothness of $\varphi^{(\ell)}$ follows from Lemma E.5. Applying the chain rule and an induction, we have

$$\dot{\varphi}^{(\ell)} = \left(\dot{\varphi} \circ \varphi^{(\ell-1)}\right)\dot{\varphi}^{(\ell-1)} = \prod_{\ell'=0}^{\ell-1} \dot{\varphi} \circ \varphi^{(\ell')}, \tag{C.23}$$

and applying the chain rule also gives

$$\ddot{\varphi}^{(\ell)} = \left(\dot{\varphi}^{(\ell-1)}\right)^2 \left(\ddot{\varphi} \circ \varphi^{(\ell-1)}\right) + \left(\ddot{\varphi}^{(\ell-1)}\right)\left(\dot{\varphi} \circ \varphi^{(\ell-1)}\right). \tag{C.24}$$

By Lemma E.5, we have $\dot{\varphi} > 0$ on $[0, \pi])$, and the formula (C.23) then implies that $\dot{\varphi}^{(\ell)} > 0$ on $[0, \pi])$ as well. Considering only angles in this half-open interval and distributing, it follows

$$\frac{\ddot{\varphi}^{(\ell)}}{\left(\dot{\varphi}^{(\ell)}\right)^2} = \frac{\ddot{\varphi} \circ \varphi^{(\ell-1)}}{\left(\dot{\varphi} \circ \varphi^{(\ell-1)}\right)^2} + \frac{1}{\dot{\varphi} \circ \varphi^{(\ell-1)}} \frac{\ddot{\varphi}^{(\ell-1)}}{\left(\dot{\varphi}^{(\ell-1)}\right)^2}$$

$$= \frac{\ddot{\varphi}}{\dot{\varphi}^2} \circ \varphi^{(\ell-1)} + \frac{1}{\dot{\varphi} \circ \varphi^{(\ell-1)}} \frac{\ddot{\varphi}^{(\ell-1)}}{\left(\dot{\varphi}^{(\ell-1)}\right)^2}.$$

Applying an induction using the previous formula and distributing in the result, we obtain

$$\frac{\ddot{\varphi}^{(\ell)}}{\left(\dot{\varphi}^{(\ell)}\right)^2} = \sum_{\ell'=0}^{\ell-1} \left( \frac{1}{\prod_{\ell''=\ell'+1}^{\ell-1} \dot{\varphi} \circ \varphi^{(\ell'')}} \right) \frac{\ddot{\varphi}}{\dot{\varphi}^2} \circ \varphi^{(\ell')}. \tag{C.25}$$

By Lemma E.5, we have $0 < \dot{\varphi} \le 1$ on $[0, \pi)$ and $\ddot{\varphi} \le 0$. Thus

$$-\frac{\ddot{\varphi}^{(\ell)}}{\left(\dot{\varphi}^{(\ell)}\right)^2} \ge -\sum_{\ell'=0}^{\ell-1} \ddot{\varphi} \circ \varphi^{(\ell')}.$$

When $\ell' > 0$, we have $\varphi^{(\ell')} \le \pi/2$, and by Lemma E.5, we have $\ddot{\varphi} \le -c < 0$ on $[0, \pi/2]$; thus, $-\ddot{\varphi} \circ \varphi^{(\ell')} \ge c$ if $\ell' > 0$. When $\ell' = 0$, we can use the fact that $\ddot{\varphi} \le 0$ on $[0, \pi]$ to get a bound $\ddot{\varphi} \le -c\mathbb{1}_{[0,\pi/2]}$. We conclude

$$-\frac{\ddot{\varphi}^{(\ell)}}{\left(\dot{\varphi}^{(\ell)}\right)^2} \ge c(\ell - 1) + c\mathbb{1}_{[0,\pi/2]}. \tag{C.26}$$

Next, we notice using the chain rule that

$$\left( \frac{1}{\dot{\varphi}^{(\ell)}} \right)' = -\frac{\ddot{\varphi}^{(\ell)}}{\left(\dot{\varphi}^{(\ell)}\right)^2},$$

and using (C.23) and Lemma E.5, we have that $\dot{\varphi}^{(\ell)}(0) = 1$. For any $\nu \in [0, \pi)$, we integrate both sides of (C.26) from $0$ to $\nu$ to obtain using the fundamental theorem of calculus

$$\frac{1}{\dot{\varphi}^{(\ell)}(\nu)} - 1 \ge c(\ell - 1)\nu + c \int_0^\nu \mathbb{1}_{[0,\pi/2]}(t) \, dt$$

$$= c(\ell - 1)\nu + c \min\{\nu, \pi/2\}$$

$$\ge \frac{c\ell\nu}{2},$$

where in the final inequality we use the inequality $\min\{\nu, \pi/2\} \ge \nu/2$, valid for $\nu \in [0, \pi]$. Rearranging, we conclude for any $0 \le \nu < \pi$

$$\dot{\varphi}^{(\ell)}(\nu) \le \frac{1}{1 + (c/2)\ell\nu},$$

and noting that the LHS of this bound is equal to $0$ at $\nu = \pi$ and the RHS is positive, we conclude the claimed bound for every $\nu \in [0, \pi]$. The second estimate claimed follows by multiplying this bound by $\nu$ on both sides, and using $1 + (c/2)\ell\nu \ge (c/2)\ell\nu$. $\qquad\square$

**Lemma C.16.** *If $\ell \in \mathbb{N}$, one has*

$$\left| \ddot{\varphi}^{(\ell)}(\nu) \right| \le \frac{C}{1 + (c/8)\ell\nu} \left( 1 + \frac{1}{(c/8)\nu} \log\left(1 + (c/8)(\ell-1)\nu\right) \right),$$

*where $C > 0$ is an absolute constant, and $c > 0$ is the absolute constant also appearing in Lemma E.5 (property 4), and in particular $c/2$ is equal to the absolute constant appearing in Lemma C.12. If $\nu \in [0, \pi]$, the RHS of this upper bound is a decreasing function of $\nu$, and moreover we have the estimates*

$$|\ddot{\varphi}^{(\ell)}| \le C\ell, \qquad \left| \nu^2 \ddot{\varphi}^{(\ell)}(\nu) \right| \le \frac{C\pi\nu}{1 + (c/8)\ell\nu} \left( 1 + \frac{8\log\ell}{c\pi} \right) \le \frac{8\pi C}{c\ell} + \frac{64C\log\ell}{c^2\ell}.$$

*Proof.* Smoothness follows from Lemma E.5; we make use of some results from the proof of Lemma C.15, in particular (C.23) and (C.25). We treat the case of $\ell = 1$ first. By Lemma E.5, we have $|\ddot{\varphi}| \leq C$ for an absolute constant $C > 0$, and since $1/(1 + (c/2)\nu) \geq 1/(3/2) = 2/3$ by the numerical estimate of the absolute constant $c > 0$ in Lemma E.5, it follows

$$|\ddot{\varphi}(\nu)| \leq \frac{3C/2}{1 + (c/2)\nu},$$

which establishes the claim when $\ell = 1$ (after worst-casing constants if necessary). Next, we assume $\ell > 1$. Multiplying both sides of (C.25) by $(\dot{\varphi}^{(\ell)})^2$ and cancelling using (C.23), we obtain

$$
\begin{aligned}
\ddot{\varphi}^{(\ell)} &= \sum_{\ell'=0}^{\ell-1} \frac{\prod_{\ell''=0}^{\ell-1} \left(\dot{\varphi} \circ \varphi^{(\ell'')}\right)^2}{\prod_{\ell''=\ell'+1}^{\ell-1} \dot{\varphi} \circ \varphi^{(\ell'')}} \frac{\ddot{\varphi} \circ \varphi^{(\ell')}}{\left(\dot{\varphi} \circ \varphi^{(\ell')}\right)^2} \\
&= \sum_{\ell'=0}^{\ell-1} \left(\prod_{\ell''=0}^{\ell'-1} \left(\dot{\varphi} \circ \varphi^{(\ell'')}\right)^2\right) \left(\prod_{\ell''=\ell'+1}^{\ell-1} \dot{\varphi} \circ \varphi^{(\ell'')}\right) \ddot{\varphi} \circ \varphi^{(\ell')} \qquad \text{(C.27)} \\
&= \dot{\varphi}^{(\ell)} \sum_{\ell'=0}^{\ell-1} \dot{\varphi}^{(\ell')} \frac{\ddot{\varphi} \circ \varphi^{(\ell')}}{\dot{\varphi} \circ \varphi^{(\ell')}},
\end{aligned}
$$

where the last equality holds at least on $[0, \pi)$, by Lemmas E.5 and C.15, and where empty products are defined to be 1. If $\ell' > 0$, we have $\varphi^{(\ell')} \leq \pi/2$, and by Lemma E.5 we have that $|\ddot{\varphi}| \leq C$ and $\dot{\varphi} \geq c' > 0$ on $[0, \pi/2]$ for absolute constants $C, C' > 0$. Separating the $\ell' = 0$ summand, this gives a bound

$$\left|\ddot{\varphi}^{(\ell)}\right| \leq C \left(\prod_{\ell'=1}^{\ell-1} \dot{\varphi} \circ \varphi^{(\ell')}\right) + \frac{C}{c'} \dot{\varphi}^{(\ell)} \sum_{\ell'=1}^{\ell-1} \dot{\varphi}^{(\ell')}. \qquad \text{(C.28)}$$

By Lemma C.15, we have $\dot{\varphi}(\nu) \leq 1/(1 + (c/2)\nu)$, and by Lemma E.5, we have $\varphi(\nu) \leq \nu$, hence $\varphi^{(\ell')}(\nu) \leq \nu$. Using concavity of $\varphi$, nonincreasingness of $\dot{\varphi}$ and nondecreasingness of $\varphi^{(\ell')}$ (which follow from Lemma E.5) and a simple re-indexing, we can write

$$
\begin{aligned}
\prod_{\ell'=1}^{\ell-1} \dot{\varphi} \circ \varphi^{(\ell')}(\nu) = \prod_{\ell'=0}^{\ell-2} \dot{\varphi} \circ \varphi^{(\ell'+1)}(\nu) &= \prod_{\ell'=0}^{\ell-2} \dot{\varphi} \circ \varphi^{(\ell')} \circ \varphi(\nu) \\
&\leq \prod_{\ell'=0}^{\ell-2} \dot{\varphi}(\varphi^{(\ell')}(\nu/2)) \\
&= \dot{\varphi}^{(\ell-1)}(\nu/2) \\
&\leq \frac{1}{1 + (c/4)(\ell-1)\nu} \\
&\leq \frac{1}{1 + (c/8)\ell\nu}
\end{aligned}
$$

where the third-to-last line follows from (C.23), the second-to-last line follows from Lemma C.15, and the last line follows from the inequality $\ell - 1 \geq \ell/2$ if $\ell \geq 2$. Following on from (C.28), we conclude by an application of Lemma C.15

$$
\begin{aligned}
\left|\ddot{\varphi}^{(\ell)}(\nu)\right| &\leq \frac{C}{1 + (c/8)\ell\nu} + \frac{C/c'}{1 + (c/2)\ell\nu} \sum_{\ell'=1}^{\ell-1} \frac{1}{1 + (c/2)\ell'\nu} \\
&\leq \frac{C}{c'} \left(\frac{1}{1 + (c/8)\ell\nu} \sum_{\ell'=0}^{\ell-1} \frac{1}{1 + (c/8)\ell'\nu}\right),
\end{aligned}
$$

where the last line simply worst-cases the constants. For any $\ell' \in \mathbb{N}_0$, the function $x \mapsto 1/(1 + (c/8)\ell'x)$ is nonincreasing, so we can estimate the sum in the previous statement using an integral,

obtaining

$$\left|\ddot{\varphi}^{(\ell)}(\nu)\right| \le \frac{C/c'}{1+(c/8)\ell\nu}\left(1+\int_0^{\ell-1}\frac{1}{1+c\nu x}\,dx\right)$$
$$\le \frac{C/c'}{1+(c/8)\ell\nu}\left(1+\frac{1}{(c/8)\nu}\log\left(1+(c/8)(\ell-1)\nu\right)\right)$$

after evaluating the integral—we define the quantity inside the parentheses on the RHS of the final inequality to be $\ell-1$ when $\nu=0$, which agrees with the integral representation in the previous line and with the unique continuous extension of the function on $(0,\pi]$ to $[0,\pi]$—which establishes the first claim.

We now move on to the study of the bound we have derived. For decreasingness, we note that the functions

$$\nu \mapsto \frac{C/c'}{1+(c/8)\ell\nu}, \qquad \nu \mapsto 1+\frac{1}{(c/8)\nu}\log\left(1+(c/8)(\ell-1)\nu\right), \qquad \text{(C.29)}$$

whose product is equal to our upper bound, are evidently both smooth nonnegative functions of $\nu$ at least on $(0,\pi]$, so that by the product rule for differentiable functions it suffices to prove that these two functions are themselves decreasing functions of $\nu$. The first function is evidently decreasing as an increasing affine reparameterization of $\nu \mapsto 1/\nu$; for the second function, after multiplying by the constant $\ell-1$ and rescaling by a positive number (when $\ell=1$, the function is identically zero on $(0,\pi]$, and the function's continuous extension as defined above equals $0$ at $0$ as well), we observe that it suffices to prove that $x \mapsto x^{-1}\log(1+x)$ is a decreasing function of $\nu$ on $(0,\infty)$. The derivative of this function is $x \mapsto (x-(1+x)\log(1+x))/(x^2(1+x))$, so it suffices to show that $x-(1+x)\log(1+x) \le 0$. Noting that the function $x \mapsto x\log x$ is convex (its second derivative is $1/x$), it follows that $x-(1+x)\log(1+x)$ is concave as a sum of concave functions, and is therefore has its graph majorized by its supporting hyperplanes; its derivative is equal to $-\log(1+x)$, which equals $0$ at $0$, and we therefore conclude from our previous reduction that the second function in (C.29) is decreasing, and that our composite upper bound is as well. For the remaining estimates, we use the concavity estimate $\log(1+x) \le x$ to obtain from our previous result

$$\left|\ddot{\varphi}^{(\ell)}(\nu)\right| \le \frac{C\ell}{1+(c/8)\ell\nu} \le C\ell,$$

since the function $x \mapsto C/(1+cx)$ is nonincreasing for any choice of the constants. Next, we use the expression we have derived in the first claim to obtain

$$\left|\nu^2\ddot{\varphi}^{(\ell)}(\nu)\right| \le \frac{C\nu}{1+(c/8)\ell\nu}\left(\nu+\frac{1}{(c/8)}\log\left(1+(c/8)(\ell-1)\nu\right)\right).$$

For any $K>0$, the function $x \mapsto x/(1+Kx)$ is nondecreasing, and using the numerical estimate $\pi(c/8)<1$ that follows from Lemma E.5, we obtain in addition $1+\pi(c/8)(\ell-1) \le \ell$ for $\ell \in \mathbb{N}$. Thus

$$\left|\nu^2\ddot{\varphi}^{(\ell)}(\nu)\right| \le \frac{C\pi^2}{1+(c/8)\ell\pi}\left(1+\frac{\log\ell}{c\pi/8}\right)$$
$$\le \frac{8\pi C}{c\ell}+\frac{64C\log\ell}{c^2\ell},$$

as claimed. □

**Lemma C.17.** *One has for every $\ell \in \{0,1,\dots,L\}$*

$$\varphi^{(\ell)}(0)=0; \qquad \dot{\varphi}^{(\ell)}(0)=1; \qquad \ddot{\varphi}^{(\ell)}(0)=-\frac{2\ell}{3\pi},$$

*and for every $\ell \in [L]$*

$$\dot{\varphi}^{(\ell)}(\pi)=\ddot{\varphi}^{(\ell)}(\pi)=0.$$

*Finally, we have $\dot{\varphi}^{(0)}(\pi)=1$ and $\ddot{\varphi}^{(0)}(\pi)=0$.*

*Proof.* The claims are consequences of Lemma E.5 when $\ell = 1$, and of $\varphi^{(0)} = \mathrm{Id}$ for smaller $\ell$; assume $\ell > 1$ below. The claim for $\varphi^{(\ell)}(0)$ follows from the fact that $\varphi(0) = 0$ and induction. For the claim about $\dot\varphi^{(\ell)}(0)$, we calculate using the chain rule

$$
\begin{aligned}
\dot\varphi^{(\ell)}(0) &= \dot\varphi(\varphi^{(\ell-1)}(0))\dot\varphi^{(\ell-1)}(0) \\
&= \dot\varphi(0)\dot\varphi^{(\ell-1)}(0) \\
&= \dot\varphi^{(\ell-1)}(0).
\end{aligned}
$$

By induction and Lemma E.5, we obtain $\dot\varphi^{(\ell)}(0) = 1$. The claim about $\dot\varphi^{(\ell)}(\pi)$ follows from the same argument. For the remaining claims about $\ddot\varphi^{(\ell)}$, we calculate using the chain rule

$$
\ddot\varphi^{(\ell)} = \left(\dot\varphi^{(\ell-1)}\right)^2 \ddot\varphi \circ \varphi^{(\ell-1)} + \left(\ddot\varphi^{(\ell-1)}\right)\dot\varphi \circ \varphi^{(\ell-1)},
$$

whence

$$
\ddot\varphi^{(\ell)}(0) = \ddot\varphi(0) + \ddot\varphi^{(\ell-1)}(0).
$$

Using Lemma E.5 to get $\ddot\varphi(0) = -2/(3\pi)$, this yields

$$
\ddot\varphi^{(\ell)}(0) = -\frac{2\ell}{3\pi}.
$$

Similarly, since we have shown $\dot\varphi^{(\ell-1)}(\pi) = 0$, we obtain $\ddot\varphi^{(\ell)}(\pi) = 0$. $\qquad\square$

**Lemma C.18.** *For first and second derivatives of $\xi^{(\ell)}$, one has*

$$
\dot\xi^{(\ell)} = -\pi^{-1}\sum_{\ell'=\ell}^{L-1}\dot\varphi^{(\ell')}\prod_{\substack{\ell''=\ell \\ \ell''\neq\ell'}}^{L-1}\left(1 - \pi^{-1}\varphi^{(\ell'')}\right),
$$

*and*

$$
\ddot\xi^{(\ell)} \tag{C.30}
$$

$$
= -\pi^{-1}\sum_{\ell'=\ell}^{L-1}\left[\ddot\varphi^{(\ell')}\prod_{\substack{\ell''=\ell \\ \ell''\neq\ell'}}^{L-1}\left(1 - \pi^{-1}\varphi^{(\ell'')}\right) - \pi^{-1}\dot\varphi^{(\ell')}\sum_{\substack{\ell''=\ell \\ \ell''\neq\ell'}}^{L-1}\dot\varphi^{(\ell'')}\prod_{\substack{\ell'''=\ell \\ \ell'''\neq\ell',\ell'''\neq\ell''}}^{L-1}\left(1 - \pi^{-1}\varphi^{(\ell''')}\right)\right], \tag{C.31}
$$

*where empty sums are interpreted as zero, and empty products as 1. In particular, one calculates*

$$
\xi^{(\ell)}(0) = 1; \qquad \dot\xi^{(\ell)}(0) = -\frac{L-\ell}{\pi}; \qquad \ddot\xi^{(\ell)}(0) = \frac{(L-\ell)(L-\ell-1)}{\pi^2} + \frac{L(L-1)-\ell(\ell-1)}{3\pi^2},
$$

*and*

$$
\xi^{(0)}(\pi) = 0; \qquad \dot\xi^{(\ell)}(\pi) = -\frac{1}{\pi}\xi^{(1)}(\pi)\mathbb{1}_{\ell=0}; \qquad \ddot\xi^{(\ell)}(\pi) = 0.
$$

*Proof.* The two derivative formulas are direct applications of the Leibniz rule to $\xi^{(\ell)}$. The claims about values at $0$ follow from plugging the results of Lemma C.17 into our derivative formulas and the definition of $\xi^{(\ell)}$. For values at $\pi$, we first note that $\varphi^{(0)}(\pi) = \pi$, from which it follows $\xi^{(0)}(\pi) = 0$. Next, we use Lemma C.17 to get that $\ddot\varphi^{(\ell)}(\pi) = 0$ for all $\ell \in \{0, 1, \dots, L\}$ and $\dot\varphi^{(\ell)}(\pi) = \mathbb{1}_{\ell=0}$ to get $\dot\xi^{(\ell)}(\pi) = -\pi^{-1}\xi^{(1)}(\pi)\mathbb{1}_{\ell=0}$. For $\ddot\xi^{(\ell)}(\pi)$, we have

$$
\begin{aligned}
\ddot\xi^{(\ell)} &= \pi^{-2}\sum_{\ell'=\ell}^{L-1}\dot\varphi^{(\ell')}(\pi)\sum_{\ell''\neq\ell'}\dot\varphi^{(\ell'')}(\pi)\prod_{\ell'''\neq\ell'',\ell'''\neq\ell'}\left(1 - \pi^{-1}\varphi^{(\ell''')}(\pi)\right) \\
&= \pi^{-2}\mathbb{1}_{\ell=0}\sum_{\ell'=1}^{L-1}\dot\varphi^{(\ell')}(\pi)\prod_{\ell''\neq\ell',\ell''\neq0}\left(1 - \pi^{-1}\varphi^{(\ell''')}(\pi)\right).
\end{aligned}
$$

If $L = 1$, the sum in the last expression is empty, and this quantity is $0$. If $L > 1$, the sum is nonempty, and every summand is equal to zero by Lemma C.17. We conclude $\ddot\xi^{(\ell)}(\pi) = 0$. $\qquad\square$

**Lemma C.19.** *If $L \geq 3$, there exists an absolute constant $0 < C \leq \pi/2$ such that on the interval $[0, C]$, one has for every $\ell = 0, 1, \ldots, L - 1$*

$$\dddot{\xi}^{(\ell)} \leq 0.$$

*Proof.* We consider functions only on $[0, \pi/2]$ in this proof. Following the calculations in the proof of Lemma C.15, we have the expression

$$\left(\varphi \circ \varphi^{(\ell-1)}\right)''' \tag{C.32}$$

$$= \left(\dot{\varphi} \circ \varphi^{(\ell-1)}\right) \dddot{\varphi}^{(\ell-1)} + 3 \left(\ddot{\varphi} \circ \varphi^{(\ell-1)}\right) \dot{\varphi}^{(\ell-1)} \ddot{\varphi}^{(\ell-1)} + \left(\dddot{\varphi} \circ \varphi^{(\ell-1)}\right) \left(\dot{\varphi}^{(\ell-1)}\right)^3. \tag{C.33}$$

Using as well Lemma E.5, we have first and second derivative estimates

$$0 \leq \dot{\varphi}^{(\ell)} \leq 1$$

and

$$-C_2 \ell \leq \ddot{\varphi}^{(\ell)} \leq -c_2 \ell.$$

By Lemma E.5, $\dddot{\varphi}$ extends to a continuous function on $[0, \pi/2]$, so in addition there exists a $\delta > 0$ such that on $[0, \delta]$ we have

$$\dddot{\varphi} \geq -\frac{1}{2\pi^2} \tag{C.34}$$

We lower bound (C.33) on $[0, \delta]$ using these estimates. For $\ell = 1$, we can do no better than (C.34). For $\ell > 1$, we can write

$$\left(\varphi \circ \varphi^{(\ell-1)}\right)''' \geq \left(\dot{\varphi} \circ \varphi^{(\ell-1)}\right) \dddot{\varphi}^{(\ell-1)} + 3\dot{\varphi}^{(\ell-1)} \left(\left(\ddot{\varphi} \circ \varphi^{(\ell-1)}\right) \ddot{\varphi}^{(\ell-1)} - \frac{1}{6\pi^2} \left(\dot{\varphi}^{(\ell-1)}\right)^2\right)$$

$$\geq \left(\dot{\varphi} \circ \varphi^{(\ell-1)}\right) \dddot{\varphi}^{(\ell-1)} + 3\dot{\varphi}^{(\ell-1)} \left(c_2^2(\ell-1) - \frac{1}{6\pi^2}\right).$$

We have the numerical estimate $c_2 = 0.14$ from Lemma E.5, and we check numerically that $(0.14)^2 > 1/6\pi^2$. This implies that on $[0, \delta]$ and for every $\ell \geq 2$, $\dddot{\varphi}^{(\ell)}$ is lower bounded by a positive number plus a scaled version of $\dddot{\varphi}^{(\ell-1)}$. We check precisely using the original formula (C.33) and Lemma E.5 for $\ell = 2$

$$\dddot{\varphi}^{(2)}(0) = 2\dddot{\varphi}(0) + 3\ddot{\varphi}(0)^2 = \frac{2}{3\pi^2} > 0,$$

so that in particular

$$\dddot{\varphi}^{(2)}(0) + \dddot{\varphi}^{(1)}(0) = \frac{1}{3\pi^2} > 0.$$

By continuity, it follows that there is a neighborhood $[0, \delta']$ on which we have $\dddot{\varphi}^{(2)} + \dddot{\varphi}^{(1)} > 0$. Thus, on $[0, \min\{\delta, \delta'\}]$, we guarantee that simultaneously

$$\dddot{\varphi}^{(\ell)} > 0 \text{ if } \ell \geq 2; \qquad \dddot{\varphi}^{(2)} + \dddot{\varphi}^{(1)} > 0.$$

Now we consider the third derivative of the skeleton summands $\xi^{(\ell)}$. Following the calculations of Lemmas C.10 and C.18, in particular applying the Leibniz rule, we observe that every term in the sum defining $\dddot{\xi}^{(\ell)}$ that does not involve a third derivative of one of the factors $(1 - (1/\pi)\varphi^{(\ell')})$ will be nonpositive, because $(1 - (1/\pi)\varphi^{(\ell')}) \geq 0$, $\dot{\varphi}^{(\ell')} \geq 0$, and $\ddot{\varphi}^{(\ell')} \leq 0$. Meanwhile, by our calculations above, on the interval $[0, \min\{\delta, \delta'\}]$, the only terms that can be positive are those with $\ell = 0$ or $\ell = 1$ where we differentiate the $\ell' = 1$ factor three times, i.e., the $\ell' = 1$ term in the sum

$$-\frac{1}{\pi} \sum_{\ell'=\ell}^{L-1} \dddot{\varphi}^{(\ell')} \prod_{\substack{\ell''=\ell \\ \ell'' \neq \ell'}}^{L-1} \left(1 - \frac{\varphi^{(\ell'')}}{\pi}\right)$$

with $\ell = 0$ or $\ell = 1$. We will compare the $\ell' = 1$ summand with the $\ell' = 2$ summand: we have that the sum of these two terms equals

$$-\frac{1}{\pi} \left( \prod_{\substack{\ell''=\ell \\ \ell'' \neq 1,2}}^{L-1} \left( 1 - \frac{\varphi^{(\ell'')}}{\pi} \right) \right) \left( \ddot{\varphi} \left( 1 - \frac{\varphi^{(2)}}{\pi} \right) + \ddot{\varphi}^{(2)} \left( 1 - \frac{\varphi}{\pi} \right) \right). \tag{C.35}$$

At 0, the quantity inside the right parentheses is equal to $\ddot{\varphi} + \ddot{\varphi}^{(2)} > 0$, by our calculations above. Thus, by continuity, there is a possibly smaller $\delta'' > 0$ such that on $[0, \delta'']$, the sum of terms (C.35) is negative. We conclude that on $[0, \min\{\delta, \delta', \delta''\}]$, we have for every $\ell \geq 0$

$$\dddot{\xi}^{(\ell)} \leq 0,$$

and since we have chosen the neighborhood sizes $\delta, \delta', \delta''$ independently of the depth $L$, we can conclude. $\qquad \square$

**Lemma C.20.** *For all $\ell \in \{0, \ldots, L-1\}$, one has*

$$\xi^{(\ell)}(\nu) \leq \frac{1 + \ell\nu/\pi}{1 + L\nu/\pi}.$$

*Proof.* We have

$$\xi^{(\ell)}(\nu) = \prod_{\ell'=\ell}^{L-1} \left( 1 - \frac{\varphi^{(\ell')}(\nu)}{\pi} \right) \leq \left( 1 - \frac{1}{\pi(L-\ell)} \sum_{\ell'=\ell}^{L-1} \varphi^{(\ell')}(\nu) \right)^{L-\ell}$$

$$\leq \exp\left( -\frac{1}{\pi} \sum_{\ell'=\ell}^{L-1} \varphi^{(\ell')}(\nu) \right), \tag{C.36}$$

where the first inequality applies the AM-GM inequality, and the second uses the standard exponential convexity estimate. Using Lemma C.13, we have

$$-\sum_{\ell'=\ell}^{L-1} \varphi^{(\ell')}(\nu) \leq -\sum_{\ell'=\ell}^{L-1} \frac{\nu}{1 + \ell'\nu/\pi} \leq -\int_{\ell}^{L} \frac{\nu}{1 + \ell'\nu/\pi} \, d\ell',$$

where the last inequality uses the fact that $\ell' \mapsto \nu/(1 + \ell'\nu/\pi)$ is nonincreasing for every $\nu \in [0, \pi]$ together with a standard estimate from the integral test. We calculate

$$\int_{\ell}^{L} \frac{\nu}{1 + \ell'\nu/\pi} \, d\ell' = \pi \log\left( \frac{1 + L\nu/\pi}{1 + \ell\nu/\pi} \right),$$

which gives the claim after substituting into (C.36). $\qquad \square$

**Lemma C.21.** *For all $\ell \in \{0, \ldots, L-1\}$, one has*

$$|\dot{\xi}^{(\ell)}(\nu)| \leq 3 \frac{L-\ell}{1 + L\nu/\pi}.$$

*Proof.* Using Lemma C.18, we have

$$\dot{\xi}^{(\ell)} = -\frac{\xi^{(1)}}{\pi} \mathbb{1}_{\ell=0} - \frac{\xi^{(\ell)}}{\pi} \sum_{\ell'=\max\{\ell,1\}}^{L-1} \frac{\dot{\varphi}^{(\ell')}}{1 - \varphi^{(\ell')}/\pi}$$

where we directly treat the case $\ell = 0$ to avoid dividing by zero at $\nu = \pi$. The triangle inequality and Lemmas E.5 and C.20 then give

$$|\dot{\xi}^{(\ell)}(\nu)| \leq \frac{2}{\pi} \left( \frac{1}{1 + L\nu/\pi} \mathbb{1}_{\ell=0} + \xi^{(\ell)}(\nu) \sum_{\ell'=\max\{\ell,1\}}^{L-1} \dot{\varphi}^{(\ell')}(\nu) \right).$$

Using Lemma C.15, we have

$$\sum_{\ell'=\ell}^{L-1} \dot{\varphi}^{(\ell')}(\nu) \le \sum_{\ell'=\ell}^{L-1} \frac{1}{1+c\ell'\nu} \le \frac{1}{1+c\ell\nu} + \int_{\ell}^{L-1} \frac{1}{1+c\ell'\nu}\, d\ell',$$

where the last inequality uses the fact that $\ell' \mapsto 1/(1+c\ell'\nu)$ is nonincreasing for every $\nu \in [0, \pi]$ together with a standard estimate from the integral test. Evaluating the integral, we obtain

$$\sum_{\ell'=\ell}^{L-1} \dot{\varphi}^{(\ell')}(\nu) \le \frac{1}{1+c\ell\nu} + \frac{1}{c\nu} \log\left(\frac{1+c(L-1)\nu}{1+c\ell\nu}\right),$$

where the second term on the RHS is defined at $\nu = 0$ by continuity. Using the standard concavity estimate $\log(1+x) \le x$, we have

$$\frac{1}{c\nu} \log\left(\frac{1+c(L-1)\nu}{1+c\ell\nu}\right) = \frac{1}{c\nu} \log\left(1 + \frac{(L-\ell-1)c\nu}{1+c\ell\nu}\right) \le \frac{L-\ell-1}{1+c\ell\nu},$$

whence

$$\sum_{\ell'=\ell}^{L-1} \dot{\varphi}^{(\ell')}(\nu) \le \frac{L-\ell}{1+c\ell\nu}. \tag{C.37}$$

Combined with the result of Lemma C.20, we conclude

$$\xi^{(\ell)}(\nu) \sum_{\ell'=\ell}^{L-1} \dot{\varphi}^{(\ell')}(\nu) \le \frac{1}{c\pi} \frac{L-\ell}{1+L\nu/\pi}.$$

The numerical estimate $c = 0.07$ in Lemma E.5 then allows us to conclude

$$|\dot{\xi}^{(\ell)}(\nu)| \le 3\frac{L-\ell}{1+L\nu/\pi},$$

as claimed. $\qquad \square$

**Lemma C.22.** *One has*

$$|\psi_1'(\nu)| \le \frac{5nL^2}{1+L\nu/\pi},$$

*and*

$$|\psi'(\nu)| \le \frac{(3/2)nL^2}{1+L\nu/\pi}.$$

*Proof.* We calculate using the chain rule

$$\psi_1' = \frac{n}{2} \sum_{\ell=0}^{L-1} \dot{\xi}^{(\ell)} \cos\varphi^{(\ell)} - \xi^{(\ell)} \dot{\varphi}^{(\ell)} \sin\varphi^{(\ell)},$$

and the triangle inequality gives

$$|\psi_1'| \le \frac{n}{2} \sum_{\ell=0}^{L-1} \left|\dot{\xi}^{(\ell)}\right| + \xi^{(\ell)} \dot{\varphi}^{(\ell)}.$$

Applying Lemmas C.15, C.20 and C.21 and Lemma E.5 to estimate the constant $c$ in Lemma C.15, we then obtain

$$
\begin{aligned}
|\psi_1'(\nu)| &\le \frac{n}{2(1+L\nu/\pi)} \sum_{\ell=0}^{L-1} 3(L-\ell) + \frac{1+\ell\nu/\pi}{1+\ell\nu/(5\pi)} \\
&\le \frac{n}{2(1+L\nu/\pi)} \sum_{\ell=0}^{L-1} 3(L-\ell) + 1 + 4\frac{\ell\nu/(5\pi)}{1+\ell\nu/(5\pi)} \\
&\le \frac{n}{2(1+L\nu/\pi)} \left(\frac{3L^2}{2} + 5L\right) \\
&\le \frac{5nL^2}{1+L\nu/\pi}.
\end{aligned}
$$

The proof of the second claim is nearly identical, since in this case we need only use the bounds on $|\dot{\xi}^{(\ell)}|$. $\qquad \square$

**Lemma C.23.** *There are absolute constants $c, C > 0$ such that for all $\ell \in \{0, \ldots, L-1\}$, one has*

$$\left| \ddot{\xi}^{(\ell)} \right| \le C \frac{L(L-\ell)(1 + \ell\nu/\pi)}{(1 + cL\nu)^2} + C \frac{(L-\ell)^2}{(1 + cL\nu)(1 + c\ell\nu)}.$$

*Proof.* By Lemmas E.5 and C.18, we can write

$$\ddot{\xi}^{(\ell)} = -\frac{\xi^{(\ell)}}{\pi} \sum_{\ell'=\max\{1,\ell\}}^{L-1} \frac{\ddot{\varphi}^{(\ell')}}{1 - \frac{\varphi^{(\ell')}}{\pi}}$$

$$+ \frac{1}{\pi^2} \left( 2\xi^{(1)} \mathbb{1}_{\ell=0} \sum_{\ell'=1}^{L-1} \frac{\dot{\varphi}^{(\ell')}}{1 - \frac{\varphi^{(\ell')}}{\pi}} + \xi^{(\ell)} \sum_{\ell'=\max\{1,\ell\}}^{L-1} \sum_{\substack{\ell''=\max\{1,\ell\} \\ \ell'' \neq \ell'}}^{L-1} \frac{\dot{\varphi}^{(\ell')} \dot{\varphi}^{(\ell'')}}{\left(1 - \frac{\varphi^{(\ell')}}{\pi}\right)\left(1 - \frac{\varphi^{(\ell'')}}{\pi}\right)} \right)$$

Focusing first on the second term, we have using Lemma E.5, (C.37) and Lemma C.20

$$2\xi^{(1)} \mathbb{1}_{\ell=0} \sum_{\ell'=1}^{L-1} \frac{\dot{\varphi}^{(\ell')}}{1 - \frac{\varphi^{(\ell')}}{\pi}} + \xi^{(\ell)} \sum_{\ell'=\max\{1,\ell\}}^{L-1} \sum_{\substack{\ell''=\max\{1,\ell\} \\ \ell'' \neq \ell'}}^{L-1} \frac{\dot{\varphi}^{(\ell')} \dot{\varphi}^{(\ell'')}}{\left(1 - \frac{\varphi^{(\ell')}}{\pi}\right)\left(1 - \frac{\varphi^{(\ell'')}}{\pi}\right)}$$

$$\le 4\xi^{(1)} \mathbb{1}_{\ell=0} \sum_{\ell'=1}^{L-1} \dot{\varphi}^{(\ell')} + 4\xi^{(\ell)} \sum_{\ell'=\max\{1,\ell\}}^{L-1} \sum_{\substack{\ell''=\max\{1,\ell\} \\ \ell'' \neq \ell'}}^{L-1} \dot{\varphi}^{(\ell')} \dot{\varphi}^{(\ell'')}.$$

We can then write using nonnegativity

$$\sum_{\ell'=\ell}^{L-1} \sum_{\substack{\ell''=\ell \\ \ell'' \neq \ell'}}^{L-1} \dot{\varphi}^{(\ell')} \dot{\varphi}^{(\ell'')} \le \sum_{\ell'=\ell}^{L-1} \sum_{\ell''=\ell}^{L-1} \dot{\varphi}^{(\ell')} \dot{\varphi}^{(\ell'')} = \left( \sum_{\ell'=\ell}^{L-1} \dot{\varphi}^{(\ell')} \right)^2,$$

and using (C.37) and Lemma C.20, we obtain thus

$$\xi^{(1)} \mathbb{1}_{\ell=0} \sum_{\ell'=1}^{L-1} \dot{\varphi}^{(\ell')} + \xi^{(\ell)} \sum_{\ell'=\max\{1,\ell\}}^{L-1} \sum_{\substack{\ell''=\max\{1,\ell\} \\ \ell'' \neq \ell'}}^{L-1} \dot{\varphi}^{(\ell')} \dot{\varphi}^{(\ell'')} \le \frac{3}{c\pi} \frac{(L-\ell)^2}{(1 + L\nu/\pi)(1 + c\ell\nu)}.$$

Regarding the first term, we have using Lemma C.16

$$\sum_{\ell'=\ell}^{L-1} |\ddot{\varphi}^{(\ell')}| \le C \sum_{\ell'=\ell}^{L-1} \frac{\ell'}{1 + (c/4)\ell'\nu} \le C \frac{L(L-\ell)}{1 + (c/4)L\nu},$$

because the function $\ell' \mapsto \ell'/(1 + c\ell'\nu)$ is nondecreasing. Applying also Lemma C.20, we obtain using the triangle inequality and worst-casing constants

$$\left| \ddot{\xi}^{(\ell)} \right| \le C_1 \frac{L(L-\ell)(1 + \ell\nu/\pi)}{(1 + cL\nu)^2} + C_2 \frac{(L-\ell)^2}{(1 + cL\nu)(1 + c\ell\nu)}.$$

$\square$

**Lemma C.24.** *One has*

$$|\psi_1''(\nu)| \le \frac{CnL^3}{1 + cL\nu},$$

*and*

$$|\psi''(\nu)| \le \frac{CnL^3}{1 + cL\nu},$$

*where $c, C > 0$ are absolute constants.*

*Proof.* We calculate using the chain rule

$$\psi_1'' = \frac{n}{2} \sum_{\ell=0}^{L-1} \ddot{\xi}^{(\ell)} \cos \varphi^{(\ell)} - 2\dot{\xi}^{(\ell)} \dot{\varphi}^{(\ell)} \sin \varphi^{(\ell)} - \xi^{(\ell)} \ddot{\varphi}^{(\ell)} \sin \varphi^{(\ell)} - \xi^{(\ell)} \left( \dot{\varphi}^{(\ell)} \right)^2 \cos \varphi^{(\ell)},$$

and the triangle inequality gives

$$|\psi_1''| \leq \frac{n}{2} \sum_{\ell=0}^{L-1} \ddot{\xi}^{(\ell)} + 2\left|\dot{\xi}^{(\ell)}\right| \dot{\varphi}^{(\ell)} + \xi^{(\ell)} \left|\ddot{\varphi}^{(\ell)}\right| + \xi^{(\ell)} \left( \dot{\varphi}^{(\ell)} \right)^2.$$

Using Lemmas C.15, C.16, C.20, C.21 and C.23 and worst-casing constants for convenience, we obtain from the last estimate

$$|\psi_1''(\nu)| \leq Cn \sum_{\ell=0}^{L-1} \left( \begin{array}{c} \frac{L(L-\ell)(1+\ell\nu/\pi)}{(1+cL\nu)^2} + \frac{(L-\ell)^2}{(1+cL\nu)(1+c\ell\nu)} \\ + \frac{L-\ell}{(1+L\nu/\pi)(1+c\ell\nu)} + \frac{1+\ell\nu/\pi}{1+L\nu/\pi} \left( \frac{1}{(1+c\ell\nu)^2} + \ell \right) \end{array} \right) \leq \frac{CnL^3}{1+cL\nu},$$

where in the second line we made some estimates along the lines of the proof of Lemma C.22 and worsened the constant $C$. The proof for $\psi$ follows from the same argument, since in this case we have the same sum of $\ddot{\xi}^{(\ell)}$ terms but none of the extra residuals. $\square$

## D CONCENTRATION AT INITIALIZATION

### D.1 NOTATION AND FRAMEWORK

We recall the expression for the neural tangent kernel, as summarized in Appendix A.5.2:

$$\Theta(\boldsymbol{x}, \boldsymbol{x}') = \left\langle \widetilde{\nabla} f_{\boldsymbol{\theta}_0}(\boldsymbol{x}), \widetilde{\nabla} f_{\boldsymbol{\theta}_0}(\boldsymbol{x}') \right\rangle$$

$$= \left\langle \boldsymbol{\alpha}^L(\boldsymbol{x}), \boldsymbol{\alpha}^L(\boldsymbol{x}') \right\rangle + \sum_{\ell=0}^{L-1} \left\langle \boldsymbol{\alpha}^\ell(\boldsymbol{x}), \boldsymbol{\alpha}^\ell(\boldsymbol{x}') \right\rangle \left\langle \boldsymbol{\beta}^\ell(\boldsymbol{x}), \boldsymbol{\beta}^\ell(\boldsymbol{x}') \right\rangle,$$

The objective of this section is to establish supporting results for the proof of Theorem B.2, which gives uniform concentration of $\Theta(\boldsymbol{x}, \boldsymbol{x}')$ over $\mathcal{M} \times \mathcal{M}$ around the deterministic skeleton kernel. We take a pointwise-uniformize approach to proving this result: Appendix D.2 establishes concentration results for the constituents of $\Theta(\boldsymbol{x}, \boldsymbol{x}')$ when $\boldsymbol{x}, \boldsymbol{x}'$ are fixed, and Appendix D.3 develops results that control the number of local support changes near points in a discretization of $\mathcal{M} \times \mathcal{M}$ in order to provide a suitable stand-in for the continuity properties necessary to uniformize these pointwise results. We collect relevant technical results and their proofs in Appendix D.4.

### D.2 POINTWISE CONCENTRATION

We fix $(\boldsymbol{x}, \boldsymbol{x}')$ in this section, and generally suppress notation involving the specific points for concision. We separate our analysis into two distinct sub-problems: "forward concentration", which consists of the study of the correlations $\langle \boldsymbol{\alpha}^\ell(\boldsymbol{x}), \boldsymbol{\alpha}^\ell(\boldsymbol{x}') \rangle$, and "backward concentration", which consists of the study of the backward feature correlations $\langle \boldsymbol{\beta}^\ell(\boldsymbol{x}), \boldsymbol{\beta}^\ell(\boldsymbol{x}') \rangle$. Forward concentration is a prerequisite of our approach to backward concentration, so we begin there.

#### D.2.1 FORWARD CONCENTRATION

**Notation.** For $\ell = 0, 1, \ldots, L$, define random variables $z_1^\ell = \|\boldsymbol{\alpha}^\ell(\boldsymbol{x})\|_2$ and $z_2^\ell = \|\boldsymbol{\alpha}^\ell(\boldsymbol{x}')\|_2$. With the convention $0 \cdot +\infty = 0$, we define for $\ell = 0, \ldots, L$, random variables $\nu^\ell$ by

$$\nu^\ell = \cos^{-1} \left( \mathbb{1}_{z_1^\ell > 0} \mathbb{1}_{z_2^\ell > 0} \left\langle \frac{\boldsymbol{\alpha}^\ell(\boldsymbol{x})}{\|\boldsymbol{\alpha}^\ell(\boldsymbol{x})\|_2}, \frac{\boldsymbol{\alpha}^\ell(\boldsymbol{x}')}{\|\boldsymbol{\alpha}^\ell(\boldsymbol{x}')\|_2} \right\rangle - \mathbb{1}_{\{z_1^\ell = 0\} \cup \{z_2^\ell = 0\}} \right).$$

These definitions guarantee that $\nu^\ell = \pi$ whenever either feature norm $z_i^\ell$ vanishes. These random variables are significant toward controlling $\Theta(\boldsymbol{x}, \boldsymbol{x}')$ because, for each $\ell$

$$\langle \boldsymbol{\alpha}^\ell(\boldsymbol{x}), \boldsymbol{\alpha}^\ell(\boldsymbol{x}') \rangle = z_1^\ell z_2^\ell \cos \nu^\ell.$$

Let us define pairs of gaussian vectors $\boldsymbol{g}_1^\ell, \boldsymbol{g}_2^\ell \sim_{\text{i.i.d.}} \mathcal{N}(\boldsymbol{0}, (2/n)\boldsymbol{I})$ that are independent of everything else in the problem. For $\ell \geq 1$, we have by rotational invariance of the Gaussian distribution and the probability chain rule

$$z_1^\ell = \left\| \left[ \boldsymbol{W}^\ell \boldsymbol{\alpha}^{\ell-1}(\boldsymbol{x}) \right]_+ \right\|_2 \overset{d}{=} \left\| \left[ \boldsymbol{g}_1^\ell \right]_+ \right\|_2 z_1^{\ell-1}.$$

Since $\boldsymbol{\alpha}^0(\boldsymbol{x}) = \boldsymbol{x}$ and $\|\boldsymbol{x}\|_2 = 1$, we have by an induction with analogous definitions

$$z_1^\ell \overset{d}{=} \prod_{\ell'=1}^\ell \left\| \left[ \boldsymbol{g}_1^{\ell'} \right]_+ \right\|_2.$$

Similarly, we have

$$z_2^\ell \overset{d}{=} \prod_{\ell'=1}^\ell \left\| \left[ \boldsymbol{g}_1^{\ell'} \right]_+ \right\|_2.$$

As for the angles, we have by rotational invariance

$$z_1^\ell z_2^\ell = \left\| \left[ \boldsymbol{W}^\ell \boldsymbol{\alpha}^{\ell-1}(\boldsymbol{x}) \right]_+ \right\|_2 \left\| \left[ \boldsymbol{W}^\ell \boldsymbol{\alpha}^{\ell-1}(\boldsymbol{x}') \right]_+ \right\|_2$$
$$\overset{d}{=} \left\| \left[ \boldsymbol{g}_1^\ell \right]_+ \right\|_2 \left\| \left[ \boldsymbol{g}_1^\ell \cos \nu^{\ell-1} + \boldsymbol{g}_2^\ell \sin \nu^{\ell-1} \right]_+ \right\|_2 z_1^{\ell-1} z_2^{\ell-1},$$

so that an inductive argument gives

$$z_1^\ell z_2^\ell \overset{d}{=} \left( \prod_{\ell'=1}^\ell \left\| \left[ \boldsymbol{g}_1^{\ell'} \right]_+ \right\|_2 \right) \left( \prod_{\ell'=1}^\ell \left\| \left[ \boldsymbol{g}_1^{\ell'} \cos \nu^{\ell'-1} + \boldsymbol{g}_2^{\ell'} \sin \nu^{\ell'-1} \right]_+ \right\|_2 \right).$$

We will write

$$\bar{z}_1^\ell = \prod_{\ell'=1}^\ell \left\| \left[ \boldsymbol{g}_1^{\ell'} \right]_+ \right\|_2, \qquad \bar{z}_2^\ell = \prod_{\ell'=1}^\ell \left\| \left[ \boldsymbol{g}_1^{\ell'} \cos \nu^{\ell'-1} + \boldsymbol{g}_2^{\ell'} \sin \nu^{\ell'-1} \right]_+ \right\|_2,$$

and similarly

$$\bar{\nu}^\ell = \cos^{-1}\left( \mathbb{1}_{\bar{z}_1^\ell \bar{z}_2^\ell > 0} \left\langle \frac{\left[ \boldsymbol{g}_1^\ell \right]_+}{\left\| \left[ \boldsymbol{g}_1^\ell \right]_+ \right\|_2}, \frac{\left[ \boldsymbol{g}_1^\ell \cos \nu^{\ell-1} + \boldsymbol{g}_2^\ell \sin \nu^{\ell-1} \right]_+}{\left\| \left[ \boldsymbol{g}_1^\ell \cos \nu^{\ell-1} + \boldsymbol{g}_2^\ell \sin \nu^{\ell-1} \right]_+ \right\|_2} \right\rangle - \mathbb{1}_{\{\bar{z}_1^\ell = 0\} \cup \{\bar{z}_2^\ell = 0\}} \right),$$

so that we obtain for the angles by a similar inductive argument

$$\nu^\ell \overset{d}{=} \bar{\nu}^\ell. \tag{D.1}$$

For technical reasons, it will be convenient to consider an auxiliary angle process, defined for $\ell \geq 1$ as

$$\hat{\nu}^\ell = \cos^{-1}\left( \mathbb{1}_{\bar{\mathcal{E}}}(\boldsymbol{g}_1^\ell, \boldsymbol{g}_2^\ell) \left\langle \frac{\left[ \boldsymbol{g}_1^\ell \right]_+}{\left\| \left[ \boldsymbol{g}_1^\ell \right]_+ \right\|_2}, \frac{\left[ \boldsymbol{g}_1^\ell \cos \hat{\nu}^{\ell-1} + \boldsymbol{g}_2^\ell \sin \hat{\nu}^{\ell-1} \right]_+}{\left\| \left[ \boldsymbol{g}_1^\ell \cos \hat{\nu}^{\ell-1} + \boldsymbol{g}_2^\ell \sin \hat{\nu}^{\ell-1} \right]_+ \right\|_2} \right\rangle \right), \tag{D.2}$$

where we define with notation from Appendix E.1

$$\bar{\mathcal{E}} = \bigcap_{i \in [n]} \left\{ (\boldsymbol{g}_1, \boldsymbol{g}_2) \,\middle|\, \forall \nu \in [0, 2\pi], \tfrac{1}{2} \leq \left\| \boldsymbol{I}_{[n]\setminus\{i\}} [\boldsymbol{g}_1 \cos \nu + \boldsymbol{g}_2 \sin \nu]_+ \right\|_2 \leq 2 \right\},$$

and $\hat{\nu}^0 = \nu^0 = \langle \boldsymbol{x}, \boldsymbol{x}' \rangle$. We then observe

$$\prod_{\ell'=1}^\ell \mathbb{1}_{\bar{\mathcal{E}}} \left( \boldsymbol{g}_1^{\ell'}, \boldsymbol{g}_2^{\ell'} \right) \leq \prod_{\ell'=1}^\ell \mathbb{1}_{\bar{z}_1^\ell \bar{z}_2^\ell > 0},$$

since the inductive structure of $\bar{z}_i^\ell$ implies that all feature norms are nonvanishing if and only if the top-level feature norms $\bar{z}_i^L$ are nonvanishing, and since the statement $\prod_{\ell'=1}^\ell \mathbb{1}_{\bar{\mathcal{E}}}(\boldsymbol{g}_1^{\ell'}, \boldsymbol{g}_2^{\ell'}) = 1$

implies by definition that $\bar{z}_1^L \geq 2^{-L}$ and $\bar{z}_2^L \geq 2^{-L}$. By Lemma E.16, as long as $n \geq 21$ the event $\bar{\mathcal{E}}$ has overwhelming probability, and in particular a union bound implies

$$\mathbb{P}\left[\prod_{\ell'=1}^{\ell} \mathbb{1}_{\bar{z}_1^{\ell} \bar{z}_2^{\ell} > 0} = 1\right] \geq \mathbb{P}\left[\prod_{\ell'=1}^{\ell} \mathbb{1}_{\bar{\mathcal{E}}}\left(g_1^{\ell'}, g_2^{\ell'}\right) = 1\right] \geq 1 - CLe^{-cn}, \tag{D.3}$$

so that

$$\mathbb{P}\left[\forall \ell = 1, 2, \ldots, L, \; \hat{\nu}^{\ell} = \bar{\nu}^{\ell}\right] \geq 1 - CLe^{-cn}. \tag{D.4}$$

We can therefore pass from $\bar{\nu}^{\ell}$ to $\hat{\nu}^{\ell}$ with negligible error.

From the expression for $\hat{\nu}^{\ell}$, we see that the angles $\hat{\nu}^0 \to \hat{\nu}^1 \to \cdots \to \hat{\nu}^L$ form a Markov chain, and we will control them using martingale techniques. For $\ell = 0, 1, \ldots, L$, we write $\mathcal{F}^{\ell}$ to denote the $\sigma$-algebra generated by the gaussian vectors $(g_1^1, g_2^1, g_1^2, g_2^2, \ldots, g_1^{\ell}, g_2^{\ell})$, so that $(\mathcal{F}^0, \ldots, \mathcal{F}^L)$ is a filtration, and the sequences of random variables $(\hat{\nu}^1, \ldots, \hat{\nu}^L)$ and $(\bar{\nu}^1, \ldots, \bar{\nu}^L)$ are adapted to $(\mathcal{F}^1, \ldots, \mathcal{F}^L)$. Moreover, with these definitions we have

$$\mathbb{E}\left[\hat{\nu}^{\ell} \mid \mathcal{F}^{\ell-1}\right] = \bar{\varphi}(\hat{\nu}^{\ell-1}),$$

where $\bar{\varphi}$ is the angle evolution function defined in Appendix E.1, which is well-approximated by the function

$$\varphi(\nu) = \cos^{-1}\left(\left(1 - \frac{\nu}{\pi}\right) + \frac{\sin \nu}{\pi}\right)$$

(see Lemmas E.1 and E.2). In the sequel, we will employ the notation $\varphi^{(\ell)}$ to denote the $\ell$-fold composition of $\varphi$ with itself. By Lemma E.5, the function $\varphi$ is smooth, and the chain rule implies the same for $\varphi^{(\ell)}$; we will employ the notation $\dot{\varphi}^{(\ell)}$ and $\ddot{\varphi}^{(\ell)}$ for the first and second derivatives of $\varphi^{(\ell)}$, respectively.

## Main results.

**Lemma D.1.** *There are absolute constants $c, C, C' > 0$ and absolute constants $K, K' > 0$ such that for any $d \geq K$, if $n \geq K' \max\{1, d^4 \log^4 n, d^3 L \log^3 n\}$ then one has for any $\ell = 1, \ldots, L$*

$$\mathbb{P}\left[\left|\langle \alpha^{\ell}(x), \alpha^{\ell}(x')\rangle - \cos \varphi^{(\ell)}(\angle(x, x'))\right| > C\sqrt{\frac{d^3 \ell \log^3 n}{n}}\right] \leq C'n^{-cd}.$$

*Proof.* We have

$$\langle \alpha^{\ell}(x), \alpha^{\ell}(x')\rangle \overset{d}{=} z_1^{\ell} z_2^{\ell} \cos \nu^{\ell},$$

and the triangle inequality (applied twice) then yields

$$\left|\langle \alpha^{\ell}(x), \alpha^{\ell}(x')\rangle - \cos \varphi^{(\ell)}(\nu^0)\right| \leq |\cos \nu^{\ell}||z_1^{\ell} z_2^{\ell} - 1| + \left|\cos \nu^{\ell} - \cos \varphi^{(\ell)}(\nu^0)\right|$$

$$\leq |z_2^{\ell}||z_1^{\ell} - 1| + |z_2^{\ell} - 1| + \left|\nu^{\ell} - \varphi^{(\ell)}(\nu^0)\right|,$$

where we also use $|\cos| \leq 1$ and that $\cos$ is 1-Lipschitz. Since $z_i^{\ell} \overset{d}{=} \bar{z}_i^{\ell}$ for $i = 1, 2$, we obtain using Lemma D.2 and the choice $n \geq KdL$

$$\mathbb{P}\left[|z_i^{\ell} - 1| > C\sqrt{\frac{d\ell}{n}}\right] \leq C'\ell e^{-d},$$

and as long as $n \geq C^2 dL$, we obtain on one of the same events

$$\mathbb{P}\left[z_2^{\ell} \leq 2\right] \geq 1 - C'\ell e^{-d}.$$

By a union bound, we obtain

$$\mathbb{P}\left[|z_2^{\ell}||z_1^{\ell} - 1| + |z_2^{\ell} - 1| \leq 3C\sqrt{\frac{d\ell}{n}}\right] \geq 1 - 2C'\ell e^{-d},$$

so that if we put $d' = d \log n$ and therefore choose $n \geq C^2 dL \log n$, we have

$$\mathbb{P}\left[|z_2^\ell||z_1^\ell - 1| + |z_2^\ell - 1| \leq 3C\sqrt{\frac{d\ell \log n}{n}}\right] \geq 1 - 2C'\ell n^{-d} \geq 1 - 2C'n^{-d},$$

with the second bound holding if $d \geq 1$ and $n \geq L$. For the remaining term, we have by the triangle inequality

$$\left|\nu^\ell - \varphi^{(\ell)}(\nu^0)\right| \leq \left|\nu^\ell - \hat{\nu}^\ell\right| + \left|\hat{\nu}^\ell - \varphi^{(\ell)}(\nu^0)\right|.$$

By (D.4), the first term on the RHS of the previous expression is equal to zero with probability at least $1 - CLe^{-cn}$ as long as $n \geq 21$. The second term can be controlled with Lemma D.3 provided we select $n, L, d$ to satisfy the hypotheses of that lemma. We thus obtain via an additional union bound

$$\mathbb{P}\left[\left|\langle \boldsymbol{\alpha}^\ell(\boldsymbol{x}), \boldsymbol{\alpha}^\ell(\boldsymbol{x}')\rangle - \cos\varphi^{(\ell)}(\nu^0)\right| > 3C\sqrt{\frac{d\ell \log n}{n}} + C'\sqrt{\frac{d^3 \log^3 n}{n\ell}}\right] \leq C''n^{-cd} + C'''\ell e^{-c'n}.$$

If $n \geq (2/c') \log L$ and $n \geq (2c/c')d \log n$, we have $C''n^{-cd} + C'''\ell e^{-c'n} \leq (C'' + C''')n^{-cd}$. The previous bound then becomes

$$\mathbb{P}\left[\left|\langle \boldsymbol{\alpha}^\ell(\boldsymbol{x}), \boldsymbol{\alpha}^\ell(\boldsymbol{x}')\rangle - \cos\varphi^{(\ell)}(\nu^0)\right| > 3C\sqrt{\frac{d\ell \log n}{n}} + C'\sqrt{\frac{d^3 \log^3 n}{n\ell}}\right] \leq (C'' + C''')n^{-cd},$$

and if we worst-case the dependence on $\ell$ and $d$ in the residual in the previous bound, we obtain

$$\mathbb{P}\left[\left|\langle \boldsymbol{\alpha}^\ell(\boldsymbol{x}), \boldsymbol{\alpha}^\ell(\boldsymbol{x}')\rangle - \cos\varphi^{(\ell)}(\nu^0)\right| > (3C + C')\sqrt{\frac{d^3\ell \log^3 n}{n}}\right] \leq (C'' + C''')n^{-cd},$$

as claimed. $\qquad\square$

**Lemma D.2.** *There are absolute constants $c, C, C' > 0$ and an absolute constant $K > 0$ such that for $i = 1, 2$, every $\ell = 1, \ldots, L$, and any $d > 0$, if $n \geq \max\{Kd\ell, 4\}$, then one has*

$$\mathbb{P}\left[|z_i^\ell - 1| > C\sqrt{\frac{d\ell}{n}}\right] \leq C'\ell e^{-cd}.$$

*Proof.* Because $z_i^\ell \overset{d}{=} \bar{z}_1^\ell$, it suffices to show

$$\mathbb{P}\left[\left|-1 + \prod_{\ell'=1}^{\ell}\left\|\left[\boldsymbol{g}_1^{\ell'}\right]_+\right\|_2\right| > C\sqrt{\frac{d\ell}{n}}\right] \leq C'\ell e^{-cd}. \tag{D.5}$$

The proof will proceed by showing concentration of the squared quantity $\prod_{\ell'=1}^{\ell}\left\|\left[\boldsymbol{g}_1^{\ell'}\right]_+\right\|_2^2$ around 1, so that we can appeal to results like Lemma D.26, and then conclude by applying an inequality for the square root to pass to the actual quantity of interest. To enter the setting of Lemma D.26, it makes sense to normalize the factors in the product by their degree, but we must avoid dividing by zero. We have $\prod_{\ell'=1}^{\ell}\left\|\left[\boldsymbol{g}_1^{\ell'}\right]_+\right\|_0 = 0$ if and only if $\prod_{\ell'=1}^{\ell}\left\|\left[\boldsymbol{g}_1^{\ell'}\right]_+\right\|_2 = 0$, and whenever $\prod_{\ell'=1}^{\ell}\left\|\left[\boldsymbol{g}_1^{\ell'}\right]_+\right\|_2 \neq 0$, we can write

$$\prod_{\ell'=1}^{\ell}\left\|\left[\boldsymbol{g}_1^{\ell'}\right]_+\right\|_2 = \left(\prod_{\ell'=1}^{\ell}\left\|\left[\boldsymbol{g}_1^{\ell'}\right]_+\right\|_2^2\right)^{1/2} \tag{D.6}$$

$$= \left(\prod_{\ell'=1}^{\ell}\frac{2}{n}\left\|\left[\boldsymbol{g}_1^{\ell'}\right]_+\right\|_0\right)^{1/2}\left(\prod_{\ell'=1}^{\ell}\frac{1}{\left\|\sqrt{\frac{n}{2}}\left[\boldsymbol{g}_1^{\ell'}\right]_+\right\|_0}\left\|\sqrt{\frac{n}{2}}\left[\boldsymbol{g}_1^{\ell'}\right]_+\right\|_2^2\right)^{1/2}, \tag{D.7}$$

using 0-homogeneity of the $\ell^0$ "norm". This leads to an extra product-of-degrees term; we will make use of Lemma D.27 to show that the product of degrees itself concentrates. We will also show that the event where a degree is zero is extremely unlikely and proceed with the degree-normalized main term by conditioning. By symmetry, the random variables $\left\|\left[g_1^\ell\right]_+\right\|_0$ are i.i.d. sums of $n$ Bernoulli random variables with rate $\frac{1}{2}$. By Lemma G.1, we then have

$$\mathbb{P}\left[\left\|\left[g_1^\ell\right]_+\right\|_0 < n/2 - t\right] \le e^{-2t^2/n},$$

and so

$$\mathbb{P}\left[\min_{\ell'=1,\dots,\ell}\left\|\left[g_1^{\ell'}\right]_+\right\|_0 < n/2 - t\right] = \mathbb{P}\left[\exists \ell' \in \{1,\dots,\ell\} : \left\|\left[g_1^{\ell'}\right]_+\right\|_0 < n/2 - t\right]$$
$$\le \ell\,\mathbb{P}\left[\left\|\left[g_1^\ell\right]_+\right\|_0 < n/2 - t\right] \le \ell e^{-2t^2/n},$$

where the first inequality applies a union bound. Putting $t = n/4$, we conclude

$$\mathbb{P}\left[\min_{\ell'=1,\dots,\ell}\left\|\left[g_1^{\ell'}\right]_+\right\|_0 < n/4\right] \le \ell e^{-n/8},$$

so that whenever $n \ge 16\log\ell$, we have $\left\|\left[g_1^{\ell'}\right]_+\right\|_0 \ge n/4$ for every $\ell' \le \ell$ with probability at least $1 - e^{-n/16}$. This gives us enough to begin working on showing concentration of the squared version of (D.5): partitioning, we can use the previous simplified bound to write

$$\mathbb{P}\left[\left|-1 + \prod_{\ell'=1}^\ell\left\|\left[g_1^{\ell'}\right]_+\right\|_2^2\right| > C\sqrt{\frac{d\ell}{n}}\right] \tag{D.8}$$

$$\le e^{-n/16} + \mathbb{P}\left[\min_{\ell'=1,\dots,\ell}\left\|\left[g_1^{\ell'}\right]_+\right\|_0 \ge n/4, \left|-1 + \prod_{\ell'=1}^\ell\left\|\left[g_1^{\ell'}\right]_+\right\|_2^2\right| > C\sqrt{\frac{d\ell}{n}}\right]. \tag{D.9}$$

Using (D.7) and the triangle inequality, we can write whenever no terms in the product vanish

$$\left|-1 + \prod_{\ell'=1}^\ell\left\|\left[g_1^{\ell'}\right]_+\right\|_2^2\right| = \left|\left(\prod_{\ell'=1}^\ell\frac{2}{n}\left\|\left[g_1^{\ell'}\right]_+\right\|_0\right)\left(\prod_{\ell'=1}^\ell\frac{1}{\left\|\sqrt{\frac{n}{2}}\left[g_1^{\ell'}\right]_+\right\|_0}\left\|\sqrt{\frac{n}{2}}\left[g_1^{\ell'}\right]_+\right\|_2^2\right) - 1\right|$$

$$\le \left|\left(\prod_{\ell'=1}^\ell\frac{2}{n}\left\|\left[g_1^{\ell'}\right]_+\right\|_0\right)\right|\left|\left(\prod_{\ell'=1}^\ell\frac{1}{\left\|\sqrt{\frac{n}{2}}\left[g_1^{\ell'}\right]_+\right\|_0}\left\|\sqrt{\frac{n}{2}}\left[g_1^{\ell'}\right]_+\right\|_2^2\right) - 1\right|$$

$$+ \left|\left(\prod_{\ell'=1}^\ell\frac{2}{n}\left\|\left[g_1^{\ell'}\right]_+\right\|_0\right) - 1\right|. \tag{D.10}$$

Moreover, we have by Lemma D.27

$$\mathbb{P}\left[\left|-1 + \prod_{\ell'=1}^\ell\frac{2}{n}\left\|\left[g_1^{\ell'}\right]_+\right\|_0\right| > 4\sqrt{\frac{d\ell}{n}}\right] \le 4\ell e^{-cd}$$

as long as $n \ge 128d\ell$. Choosing in addition $n \ge 4d\ell$ and using nonnegativity, this implies

$$\mathbb{P}\left[\left|\prod_{\ell'=1}^\ell\frac{2}{n}\left\|\left[g_1^{\ell'}\right]_+\right\|_0\right| > 2\right] \le 4\ell e^{-cd},$$

occurring on the same event. Combining the previous two bounds with (D.10) and (D.9) via another partition, we get

$$\mathbb{P}\left[\left|-1 + \prod_{\ell'=1}^\ell\left\|\left[g_1^{\ell'}\right]_+\right\|_2^2\right| > C\sqrt{\frac{d\ell}{n}}\right] \le e^{-n/16} + 4\ell e^{-cd}$$

$$+ \mathbb{P}\left[\begin{array}{c}\min_{\ell'=1,\dots,\ell}\left\|\left[g_1^{\ell'}\right]_+\right\|_0 \ge n/4, \\ \left|-1 + \prod_{\ell'=1}^\ell\frac{1}{\left\|\sqrt{\frac{n}{2}}\left[g_1^{\ell'}\right]_+\right\|_0}\left\|\sqrt{\frac{n}{2}}\left[g_1^{\ell'}\right]_+\right\|_2^2\right| > (C/2 + 2)\sqrt{\frac{d\ell}{n}}\end{array}\right], \tag{D.11}$$

where we use here that on the event $\{\min_{\ell'=1,\ldots,\ell}\|[g_1^{\ell'}]_+\|_0 \geq n/4\}$, the quantity $\prod_{\ell'=1}^{\ell}\left\|[g_1^{\ell'}]_+\right\|_2$ is nonzero almost surely, which allowed us to invoke the identities (D.7). For $(k_1,\ldots,k_\ell) \in [n]^\ell$, we define events $\mathcal{E}_1^{k_1},\ldots,\mathcal{E}_\ell^{k_\ell}$ by

$$\mathcal{E}_{\ell'}^{k_{\ell'}} = \left\{\left\|\sqrt{\frac{n}{2}}[g_1^{\ell'}]_+\right\|_0 = k_{\ell'}\right\}.$$

Conditioning and then relaxing the bounds, we can write

$$\mathbb{P}\left[\min_{\ell'=1,\ldots,\ell}\left\|[g_1^{\ell'}]_+\right\|_0 \geq n/4, \left|-1 + \prod_{\ell'=1}^{\ell}\frac{1}{\left\|\sqrt{\frac{n}{2}}[g_1^{\ell'}]_+\right\|_0}\left\|\sqrt{\frac{n}{2}}[g_1^{\ell'}]_+\right\|_2^2\right| > C'\sqrt{\frac{d\ell}{n}}\right]$$

$$\leq \sum_{\substack{(k_1,\ldots,k_\ell)\in[n]^\ell \\ k_{\ell'} \geq \lceil n/4 \rceil}} \frac{\mathbb{P}\left[\left|-1 + \prod_{\ell'=1}^{\ell}\frac{1}{\left\|\sqrt{\frac{n}{2}}[g_1^{\ell'}]_+\right\|_0}\left\|\sqrt{\frac{n}{2}}[g_1^{\ell'}]_+\right\|_2^2\right| > C'\sqrt{\frac{d\ell}{n}} \;\middle|\; \mathcal{E}_1^{k_1},\ldots,\mathcal{E}_\ell^{k_\ell}\right]}{*\mathbb{P}\left[\mathcal{E}_1^{k_1},\ldots,\mathcal{E}_\ell^{k_\ell}\right]}.$$

Conditioned on $\mathcal{E}_1^{k_1},\ldots,\mathcal{E}_\ell^{k_\ell}$ with $k_{\ell'} > 0$, the random variable $\prod_{\ell'=1}^{\ell}\|\sqrt{\frac{n}{2}}[g_1^{\ell'}]_+\|_2^2/\|\sqrt{\frac{n}{2}}[g_1^{\ell'}]_+\|_0$ is distributed as a product of independent degree-normalized standard $\chi^2$ random variables with minimum degree $\min\{k_1,\ldots,k_\ell\}$. An application of Lemma D.26 then yields immediately

$$\mathbb{P}\left[\left|-1 + \prod_{\ell'=1}^{\ell}\frac{1}{\left\|\sqrt{\frac{n}{2}}[g_1^{\ell'}]_+\right\|_0}\left\|\sqrt{\frac{n}{2}}[g_1^{\ell'}]_+\right\|_2^2\right| > C'\sqrt{\frac{d\ell}{n}} \;\middle|\; \mathcal{E}_1^{k_1},\ldots,\mathcal{E}_\ell^{k_\ell}\right] \leq C''\ell e^{-cd}$$

as long as $n \geq K''d\ell$, whence

$$\mathbb{P}\left[\min_{\ell'=1,\ldots,\ell}\left\|[g_1^{\ell'}]_+\right\|_0 \geq n/4, \left|-1 + \prod_{\ell'=1}^{\ell}\frac{1}{\left\|\sqrt{\frac{n}{2}}[g_1^{\ell'}]_+\right\|_0}\left\|\sqrt{\frac{n}{2}}[g_1^{\ell'}]_+\right\|_2^2\right| > C'\sqrt{\frac{d\ell}{n}}\right]$$

$$\leq C''\ell e^{-cd}.$$

Combining this previous bound with (D.11) yields

$$\mathbb{P}\left[\left|-1 + \prod_{\ell'=1}^{\ell}\left\|[g_1^{\ell'}]_+\right\|_2^2\right| > C\sqrt{\frac{d\ell}{n}}\right] \leq e^{-n/16} + C'\ell e^{-cd},$$

where we worst-cased constants in the probability bound. If we choose $n \geq 4C^2 d\ell$, we have $C\sqrt{d\ell/n} \leq 1/2$, and we obtain on the event in the previous bound

$$\mathbb{P}\left[\left|-1 + \prod_{\ell'=1}^{\ell}\left\|[g_1^{\ell'}]_+\right\|_2^2\right| > \frac{1}{2}\right] \leq e^{-n/16} + C'\ell e^{-cd}.$$

In particular, on the complement of the event in the previous bound, the product lies in $[1/2, 3/2]$. To conclude, we can linearize the square root near 1 to obtain an analogous bound for the product of the norms. Taylor expansion of the smooth function $x \mapsto x^{1/2}$ about the point 1 gives

$$\sqrt{x} - 1 = \frac{1}{2}(x-1) - \frac{1}{8}k^{-3/2}(x-1)^2,$$

where $k$ lies between $x$ and 1. In particular, if $x \geq \frac{1}{2}$, we have

$$\frac{1}{2}(x-1) - \frac{1}{\sqrt{2}}(x-1)^2 \leq \sqrt{x} - 1 \leq \frac{1}{2}(x-1),$$

so that

$$\left|(\sqrt{x} - 1) - \frac{1}{2}(x-1)\right| \leq \frac{1}{\sqrt{2}}(x-1)^2.$$

Thus, when $x \geq \frac{1}{2}$ we have by the triangle inequality

$$\left| \sqrt{x} - 1 \right| \leq \frac{1}{\sqrt{2}} \left( x - 1 \right)^2 + \frac{1}{2} |x - 1|.$$

from which we conclude based on a partition and our previous choices of large $n$

$$\mathbb{P}\left[ \left| -1 + \prod_{\ell'=1}^{\ell} \left\| \left[ \boldsymbol{g}_1^{\ell'} \right]_+ \right\|_2 \right| > 2C\sqrt{\frac{d\ell}{n}} \right] \leq 2e^{-n/16} + 2C'\ell e^{-cd},$$

which yields the claimed probability bound when $n \geq 16d$. □

**Lemma D.3.** *There are absolute constants $c, C, C_0 > 0$ and absolute constants $K, K' > 0$ such that for any $L_{\max} \in \mathbb{N}$ and any $d \geq K$, if $n \geq K' \max\{1, d^4 \log^4 n, d^3 L_{\max} \log^3 n\}$, then one has*

$$\mathbb{P}\left[ \exists L \in [L_{\max}] : \left| \hat{\nu}^L - \varphi^{(L)}(\nu^0) \right| > C_0 \sqrt{\frac{d^3 \log^3 n}{nL}} \right] \leq Cn^{-cd}. \qquad \text{(D.12)}$$

*Proof.* The proof uses a recursive construction involving $L \in [L_{\max}]$. Before beginning the main argument, we will define the key quantities that appear and enforce bounds on the parameters to obtain certain estimates. For each $L \in [L_{\max}]$, we define the event

$$\mathcal{E}_L = \left\{ \left| \hat{\nu}^L - \varphi^{(L)}(\nu^0) \right| > C_0 \sqrt{\frac{d^3 \log^3 n}{nL}} \right\},$$

where $C_0 > 0$ is an absolute constant whose value we will specify below, so that $\mathcal{E}_L \in \mathcal{F}^L$, and our task is to produce an appropriate measure bound on $\bigcup_{L \in [L_{\max}]} \mathcal{E}_L$. For notational convenience, we also define $\mathcal{E}_0 = \varnothing$. For each $L \in [L_{\max}]$ and each $\ell \in [L]$, we define

$$\Delta_L^\ell = \varphi^{(L-\ell)}(\hat{\nu}^\ell) - \varphi^{(L-\ell+1)}(\hat{\nu}^{\ell-1}),$$

so that for every $L$, $\Delta_L^1, \ldots, \Delta_L^L$ is adapted to the sequence $\mathcal{F}^1, \ldots, \mathcal{F}^L$, and we have the decomposition

$$\hat{\nu}^L - \varphi^{(L)}(\nu^0) = \sum_{\ell=1}^{L} \Delta_L^\ell.$$

In particular, we have

$$\mathcal{E}_L = \left\{ \left| \sum_{\ell=1}^{L} \Delta_L^\ell \right| > C_0 \sqrt{\frac{d^3 \log^3 n}{nL}} \right\}.$$

The sequences $(\Delta_L^\ell)_{\ell \in L}$ are not quite martingale difference sequences, but we will show they are very nearly so: writing

$$\Delta_L^\ell = \underbrace{\left( \Delta_L^\ell - \mathbb{E}\left[ \Delta_L^\ell \mid \mathcal{F}^{\ell-1} \right] \right)}_{\bar{\Delta}_L^\ell} + \mathbb{E}\left[ \Delta_L^\ell \mid \mathcal{F}^{\ell-1} \right],$$

we have that $(\bar{\Delta}_L^\ell)_{\ell \in L}$ is a martingale difference sequence, which can be controlled using truncation and martingale techniques, and the extra conditional expectation term can be controlled analytically. In particular, we have the following estimates: by Lemma D.24, we have if $n \geq \max\{K_1 \log^4 n, K_2 L_{\max}\}$ that for every $L \in [L_{\max}]$ and every $\ell \in [L]$

$$\left| \mathbb{E}\left[ \Delta_L^\ell \mid \mathcal{F}^{\ell-1} \right] \right| \leq C_1 \frac{\log n}{n} \frac{\hat{\nu}^{\ell-1}}{1 + (c_0/64)(L-\ell)\hat{\nu}^{\ell-1}} (1 + \log L) + C_2 \frac{1}{n^2}; \qquad \text{(D.13)}$$

by the first result in Lemma D.25 we have for every $d \geq \max\{K_3, 6/c_1\}$ that if $n \geq K_4 d^4 \log^4 n$, then for every $L \in [L_{\max}]$ and every $\ell \in [L]$ (and after worsening constants)

$$\mathbb{P}\left[ \left| \Delta_L^\ell \right| > 2C_3 \sqrt{\frac{d \log n}{n}} \frac{\hat{\nu}^{\ell-1}}{1 + (c_0/64)(L-\ell)\hat{\nu}^{\ell-1}} + \frac{2C_2}{n^2} \mid \mathcal{F}^{\ell-1} \right] \leq C_5 n^{-c_1 d}; \qquad \text{(D.14)}$$

and by the second result in Lemma D.25, we have by our previous choices of $n$, $d$, and $L_{\max}$ that for every $L \in [L_{\max}]$ and every $\ell \in [L]$ (after worsening constants)

$$\mathbb{E}\left[\left(\Delta_L^\ell\right)^2 \mid \mathcal{F}^{\ell-1}\right] \le 4C_3^2 \frac{d \log n}{n} \left(\frac{\hat\nu^{\ell-1}}{1 + (c_0/64)(L - \ell)\hat\nu^{\ell-1}}\right)^2 + \frac{C_4}{n^4}. \tag{D.15}$$

The main line of the argument will consist of showing that a measure bound of the form (D.12) on $\bigcup_{\ell \in [L-1]} \mathcal{E}_\ell$ implies one of the same form on $\bigcup_{\ell \in [L]} \mathcal{E}_\ell$. For any $L \in [L_{\max}]$, on the event $\mathcal{E}_L^{\mathsf{c}}$ we have

$$\hat\nu^L \le \varphi^{(L)}(\hat\nu^0) + C_0 \sqrt{\frac{d^3 \log^3 n}{nL}}$$

$$\le \frac{2}{c_0 L} + C_0 \sqrt{\frac{d^3 \log^3 n}{nL}} \tag{D.16}$$

$$\le \frac{3}{c_0 L},$$

where the second inequality follows from Lemma C.12, and the third follows from the choice $n \ge (C_0 c_0)^2 d^3 L \log^3 n$. In particular, if we make the choice $n \ge (C_0 c_0)^2 d^3 L_{\max} \log^3 n$, we have (D.16) on $\mathcal{E}_L^{\mathsf{c}}$ for every $L \in [L_{\max}]$. Accordingly, for every $L \in [L_{\max}]$ and every $\ell \in [L]$ we define truncation events $\mathcal{G}_L^\ell$ by

$$\mathcal{G}_L^\ell = \left\{ |\Delta_L^\ell| \le 2C_3 \sqrt{\frac{d \log n}{n}} \frac{\hat\nu^{\ell-1}}{1 + (c_0/64)(L - \ell)\hat\nu^{\ell-1}} + \frac{2C_2}{n^2} \right\} \cap \mathcal{E}_{\ell-1}^{\mathsf{c}}. \tag{D.17}$$

We have $\mathcal{G}_L^\ell \in \mathcal{F}^\ell$, and a union bound and (D.14) imply

$$\mathbb{P}\left[\left(\bigcap_{\ell \in [L]} \mathcal{G}_L^\ell\right)^{\mathsf{c}} \middle| \mathcal{F}^{L-1}\right] \le C_5 L n^{-c_1 d} + \mathbb{P}\left[\bigcup_{\ell' \in [L-1]} \mathcal{E}_{\ell'} \middle| \mathcal{F}^{L-1}\right]$$

$$= C_5 L n^{-c_1 d} + \mathbb{1}_{\bigcup_{\ell' \in [L-1]} \mathcal{E}_{\ell'}},$$

where the second line uses the fact that $\mathcal{E}_{\ell'} \in \mathcal{F}^{\ell'}$. In particular, taking expectations recovers

$$\mathbb{P}\left[\left(\bigcap_{\ell \in [L]} \mathcal{G}_L^\ell\right)^{\mathsf{c}}\right] \le C_5 L n^{-c_1 d} + \mathbb{P}\left[\bigcup_{\ell' \in [L-1]} \mathcal{E}_{\ell'}\right]. \tag{D.18}$$

In addition, by (D.16) we have on $\mathcal{E}_{\ell-1}^{\mathsf{c}}$

$$\frac{\hat\nu^{\ell-1}}{1 + (c_0/64)(L - \ell)\hat\nu^{\ell-1}} \le \frac{3}{c_0} \frac{1}{(\ell - 1) + (3/64)(L - \ell)}$$

$$= \frac{3}{c_0} \frac{1}{(3/64)L + (61/64)\ell - 1}$$

$$\le \frac{64}{c_0(L - 1)}$$

$$\le \frac{128}{c_0 L},$$

where the final inequality requires $L \ge 2$. Thus, when $L \ge 2$, we have on $\mathcal{G}_L^\ell$ that

$$|\Delta_L^\ell| \le \frac{256 C_3}{c_0 L} \sqrt{\frac{d \log n}{n}} + \frac{2C_2}{n^2}$$

$$\le \underbrace{\frac{512 C_3}{c_0}}_{2K_0} \sqrt{\frac{d \log n}{nL^2}}, \tag{D.19}$$

where the final inequality holds when $d \geq 1$ and $n \geq (C_2 c_0 / 128 C_3)^{2/3} L^{2/3}$. Similarly, when $L \geq 2$, on $\mathcal{E}_{\ell-1}^{\mathsf{c}}$ we have by (D.15)

$$
\begin{aligned}
\mathbb{E}\left[\left(\Delta_L^\ell\right)^2 \mid \mathcal{F}^{\ell-1}\right] &\leq \frac{2^{16} C_3^2}{c_0^2} \frac{d \log n}{nL^2} + \frac{C_4}{n^4} \\
&\leq \frac{2^{17} C_3^2}{c_0^2} \frac{d \log n}{nL^2} = 2K_0^2 \frac{d \log n}{nL^2},
\end{aligned}
\tag{D.20}
$$

where the second inequality holds when $d \geq 1$ and $n \geq (C_4 c_0^2 / 2^{17} C_3^2)^{1/3} L^{2/3}$; and in the same setting we have by (D.13)

$$
\begin{aligned}
\left|\mathbb{E}\left[\Delta_L^\ell \mid \mathcal{F}^{\ell-1}\right]\right| &\leq \frac{128 C_1}{c_0} \frac{(1 + \log L) \log n}{nL} + \frac{C_2}{n^2} \\
&\leq \frac{256 C_1}{c_0} \frac{(1 + \log L) \log n}{nL},
\end{aligned}
\tag{D.21}
$$

where the second inequality holds when $n \geq (C_2 c_0 / 128 C_1) L$. In particular, if we enforce these conditions with $L_{\max}$ in place of $L$, we have that (D.19) to (D.21) hold for all $2 \leq L \leq L_{\max}$ (with (D.20) and (D.21) holding on $\mathcal{E}_{\ell-1}^{\mathsf{c}}$).

We begin the recursive construction. We will enforce $C_0 = \max\{4\pi C_3, 6K_0\}$ for the absolute constant in the definition of $\mathcal{E}_\ell$. The main tool is the elementary identity

$$
\mathbb{P}\left[\bigcup_{\ell \in [L]} \mathcal{E}_\ell\right] = \mathbb{P}\left[\bigcup_{\ell \in [L-1]} \mathcal{E}_\ell\right] + \mathbb{P}\left[\mathcal{E}_L \cap \bigcap_{\ell \in [L-1]} \mathcal{E}_\ell^{\mathsf{c}}\right],
\tag{D.22}
$$

which allows us to leverage an inductive argument provided we can control $\mathbb{P}[\mathcal{E}_L \cap \bigcap_{\ell \in [L-1]} \mathcal{E}_\ell^{\mathsf{c}}]$, the probability that the $L$-th angle deviates above its nominal value subject to all prior angles being controlled. The case $L = 1$ can be addressed directly: (D.14) gives

$$
\mathbb{P}\left[\left|\Delta_1^1\right| > 2\pi C_3 \sqrt{\frac{d \log n}{n}} + \frac{2C_2}{n^2}\right] \leq C_5 n^{-c_1 d},
$$

and as long as $d \geq 1$ and $n \geq (C_2 / \pi C_3)^{2/3}$, this implies

$$
\mathbb{P}\left[\left|\Delta_1^1\right| > 4\pi C_3 \sqrt{\frac{d \log n}{n}}\right] \leq C_5 n^{-c_1 d}.
\tag{D.23}
$$

This gives a suitable measure bound on $\mathcal{E}_1$, after choosing $d \geq 1$ and $n \geq e$ so that $d^3 \log^3 n \geq d \log n$. We now assume $L \geq 2$. By the triangle inequality, we have

$$
\left|\sum_{\ell=1}^L \Delta_L^\ell\right| \leq \left|\sum_{\ell=1}^L \bar{\Delta}_L^\ell\right| + \sum_{\ell=1}^L \left|\mathbb{E}\left[\Delta_L^\ell \mid \mathcal{F}^{\ell-1}\right]\right|,
\tag{D.24}
$$

and we therefore have for any $t > 0$

$$
\begin{aligned}
&\mathbb{P}\left[\left\{\left|\sum_{\ell=1}^L \Delta_L^\ell\right| > t\right\} \cap \bigcap_{\ell \in [L-1]} \mathcal{E}_\ell^{\mathsf{c}}\right] \\
&\leq \mathbb{P}\left[\left\{\left|\sum_{\ell=1}^L \bar{\Delta}_L^\ell\right| + \sum_{\ell=1}^L \left|\mathbb{E}\left[\Delta_L^\ell \mid \mathcal{F}^{\ell-1}\right]\right| > t\right\} \cap \bigcap_{\ell \in [L-1]} \mathcal{E}_\ell^{\mathsf{c}}\right] \\
&= \mathbb{P}\left[\mathbb{1}_{\bigcap_{\ell \in [L-1]} \mathcal{E}_\ell^{\mathsf{c}}} \left|\sum_{\ell=1}^L \bar{\Delta}_L^\ell\right| + \mathbb{1}_{\bigcap_{\ell \in [L-1]} \mathcal{E}_\ell^{\mathsf{c}}} \sum_{\ell=1}^L \left|\mathbb{E}\left[\Delta_L^\ell \mid \mathcal{F}^{\ell-1}\right]\right| > t\right].
\end{aligned}
\tag{D.25}
$$

By (D.21), we have

$$
\mathbb{1}_{\bigcap_{\ell \in [L-1]} \mathcal{E}_\ell^{\mathsf{c}}} \sum_{\ell=1}^L \left|\mathbb{E}\left[\Delta_L^\ell \mid \mathcal{F}^{\ell-1}\right]\right| \leq \frac{256 C_1}{c_0} \frac{(1 + \log L) \log n}{n}.
\tag{D.26}
$$

For the remaining term, we have by the triangle inequality

$$\left| \sum_{\ell=1}^{L} \bar{\Delta}_L^{\ell} \right| \leq \begin{array}{c} \left| \sum_{\ell=1}^{L} \Delta_L^{\ell} - \Delta_L^{\ell} \mathbb{1}_{\mathcal{G}_L^{\ell}} \right| + \left| \sum_{\ell=1}^{L} \Delta_L^{\ell} \mathbb{1}_{\mathcal{G}_L^{\ell}} - \mathbb{E}\left[ \Delta_L^{\ell} \mathbb{1}_{\mathcal{G}_L^{\ell}} \mid \mathcal{F}^{\ell-1} \right] \right| \\ + \left| \sum_{\ell=1}^{L} \mathbb{E}\left[ \Delta_L^{\ell} \mathbb{1}_{\mathcal{G}_L^{\ell}} \mid \mathcal{F}^{\ell-1} \right] - \mathbb{E}\left[ \Delta_L^{\ell} \mid \mathcal{F}^{\ell-1} \right] \right|. \end{array} \tag{D.27}$$

By (D.17), an integration of (D.14), and a union bound, we have

$$\mathbb{P}\left[ \mathbb{1}_{\bigcap_{\ell \in [L-1]} \mathcal{E}_\ell^c} \left| \sum_{\ell=1}^{L} \Delta_L^{\ell} - \Delta_L^{\ell} \mathbb{1}_{\mathcal{G}_L^{\ell}} \right| > 0 \right]$$

$$\leq \mathbb{P}\left[ \bigcup_{\ell \in [L]} \left\{ |\Delta_L^{\ell}| > 2C_3 \sqrt{\frac{d \log n}{n}} \frac{\hat{\nu}^{\ell-1}}{1 + (c_0/64)(L-\ell)\hat{\nu}^{\ell-1}} + \frac{2C_2}{n^2} \right\} \right] \tag{D.28}$$

$$\leq C_5 L n^{-c_1 d},$$

and we have

$$\left| \sum_{\ell=1}^{L} \mathbb{E}\left[ \Delta_L^{\ell} \mathbb{1}_{\mathcal{G}_L^{\ell}} \mid \mathcal{F}^{\ell-1} \right] - \mathbb{E}\left[ \Delta_L^{\ell} \mid \mathcal{F}^{\ell-1} \right] \right|$$

$$\leq \sum_{\ell=1}^{L} \mathbb{E}\left[ |\Delta_L^{\ell}| \mathbb{1}_{(\mathcal{G}_L^{\ell})^c} \mid \mathcal{F}^{\ell-1} \right]$$

$$\leq \pi \sum_{\ell=1}^{L} \mathbb{P}\left[ (\mathcal{G}_L^{\ell})^c \mid \mathcal{F}^{\ell-1} \right]$$

$$\leq \pi \sum_{\ell=1}^{L} \mathbb{P}\left[ \left\{ |\Delta_L^{\ell}| > 2C_3 \sqrt{\frac{d \log n}{n}} \frac{\hat{\nu}^{\ell-1}}{1 + (c_0/64)(L-\ell)\hat{\nu}^{\ell-1}} + \frac{2C_2}{n^2} \right\} \cup \mathcal{E}_{\ell-1} \mid \mathcal{F}^{\ell-1} \right]$$

$$\leq \pi C_5 L n^{-c_1 d} + \pi \sum_{\ell=1}^{L-1} \mathbb{1}_{\mathcal{E}_\ell},$$

where the first line uses linearity of the conditional expectation and the triangle inequality for sums and for the integral; the second line uses the worst-case bound of $\pi$ on the magnitude of the increments $\Delta_L^{\ell}$; the third line uses (D.17); and the fourth line uses a union bound, $\mathcal{E}_{\ell-1} \in \mathcal{F}^{\ell-1}$, and (D.14). Multiplying both sides of the final bound by $\mathbb{1}_{\bigcap_{\ell \in [L-1]} \mathcal{E}_\ell^c}$, we conclude

$$\mathbb{1}_{\bigcap_{\ell \in [L-1]} \mathcal{E}_\ell^c} \left| \sum_{\ell=1}^{L} \mathbb{E}\left[ \Delta_L^{\ell} \mathbb{1}_{\mathcal{G}_L^{\ell}} \mid \mathcal{F}^{\ell-1} \right] - \mathbb{E}\left[ \Delta_L^{\ell} \mid \mathcal{F}^{\ell-1} \right] \right| \leq \mathbb{1}_{\bigcap_{\ell \in [L-1]} \mathcal{E}_\ell^c} \pi C_5 L n^{-c_1 d} \leq \pi C_5 L n^{-c_1 d}. \tag{D.29}$$

For the remaining term in (D.27), we first observe

$$\mathbb{E}\left[ \left( \Delta_L^{\ell} \mathbb{1}_{\mathcal{G}_L^{\ell}} - \mathbb{E}\left[ \Delta_L^{\ell} \mathbb{1}_{\mathcal{G}_L^{\ell}} \mid \mathcal{F}^{\ell-1} \right] \right)^2 \mid \mathcal{F}^{\ell-1} \right] \leq \mathbb{E}\left[ (\Delta_L^{\ell})^2 \mathbb{1}_{\mathcal{G}_L^{\ell}} \mid \mathcal{F}^{\ell-1} \right]$$

$$\leq \mathbb{E}\left[ (\Delta_L^{\ell})^2 \mid \mathcal{F}^{\ell-1} \right],$$

where the first line uses the centering property of the $L^2$ norm, and the second line uses $(\Delta_L^{\ell})^2 \geq 0$ to drop the indicator for $\mathcal{G}_L^{\ell}$. For notational simplicity, we define

$$V^L = \sum_{\ell=1}^{L} \mathbb{E}\left[ \left( \Delta_L^{\ell} \mathbb{1}_{\mathcal{G}_L^{\ell}} - \mathbb{E}\left[ \Delta_L^{\ell} \mathbb{1}_{\mathcal{G}_L^{\ell}} \mid \mathcal{F}^{\ell-1} \right] \right)^2 \mid \mathcal{F}^{\ell-1} \right],$$

so that our previous bound and (D.20) imply

$$\bigcap_{\ell \in [L-1]} \mathcal{E}_\ell^c \subset \left\{ V^L \leq 2K_0^2 \frac{d \log n}{nL} \right\}.$$

This implies that for any $t > 0$

$$\mathbb{P}\left[\mathbb{1}_{\bigcap_{\ell\in[L-1]}\mathcal{E}_\ell^c}\left|\sum_{\ell=1}^L \Delta_L^\ell \mathbb{1}_{\mathcal{G}_L^\ell} - \mathbb{E}\left[\Delta_L^\ell \mathbb{1}_{\mathcal{G}_L^\ell}\;\Big|\;\mathcal{F}^{\ell-1}\right]\right| > t\right]$$

$$= \mathbb{P}\left[\left\{\bigcap_{\ell\in[L-1]}\mathcal{E}_\ell^c\right\}\cap\left\{\left|\sum_{\ell=1}^L \Delta_L^\ell \mathbb{1}_{\mathcal{G}_L^\ell} - \mathbb{E}\left[\Delta_L^\ell \mathbb{1}_{\mathcal{G}_L^\ell}\;\Big|\;\mathcal{F}^{\ell-1}\right]\right| > t\right\}\right]$$

$$\leq \mathbb{P}\left[\left\{V^L \leq 2K_0^2 \frac{d\log n}{nL}\right\}\cap\left\{\left|\sum_{\ell=1}^L \Delta_L^\ell \mathbb{1}_{\mathcal{G}_L^\ell} - \mathbb{E}\left[\Delta_L^\ell \mathbb{1}_{\mathcal{G}_L^\ell}\;\Big|\;\mathcal{F}^{\ell-1}\right]\right| > t\right\}\right].$$

The previous term can be controlled using Lemma G.5 and (D.19):

$$\mathbb{P}\left[\left\{V^L \leq 2K_0^2 \frac{d\log n}{nL}\right\}\cap\left\{\left|\sum_{\ell=1}^L \Delta_L^\ell \mathbb{1}_{\mathcal{G}_L^\ell} - \mathbb{E}\left[\Delta_L^\ell \mathbb{1}_{\mathcal{G}_L^\ell}\;\Big|\;\mathcal{F}^{\ell-1}\right]\right| > t\right\}\right]$$

$$\leq 2\exp\left(-\frac{t^2/2}{2K_0^2\frac{d\log n}{nL} + (2K_0/3)t\sqrt{\frac{d\log n}{nL^2}}}\right).$$

Setting $t = 3K_0\sqrt{d^3\log^3 n/nL}$, we obtain

$$\mathbb{P}\left[\mathbb{1}_{\bigcap_{\ell\in[L-1]}\mathcal{E}_\ell^c}\left|\sum_{\ell=1}^L \Delta_L^\ell \mathbb{1}_{\mathcal{G}_L^\ell} - \mathbb{E}\left[\Delta_L^\ell \mathbb{1}_{\mathcal{G}_L^\ell}\;\Big|\;\mathcal{F}^{\ell-1}\right]\right| > 3K_0\sqrt{\frac{d^3\log^3 n}{nL}}\right]$$

$$\leq 2\exp\left(-\frac{9}{4}\frac{d^2\log^2 n}{1 + \frac{d\log n}{\sqrt{L}}}\right) \tag{D.30}$$

$$\leq 2n^{-(9/8)d},$$

where the last line uses the bounds $L \geq 1$ and $d\log n/(1 + d\log n) \geq \frac{1}{2}$ if $d \geq 1$ and $n \geq e$. Combining (D.28) to (D.30) in (D.27) via a union bound, we obtain

$$\mathbb{P}\left[\mathbb{1}_{\bigcap_{\ell\in[L-1]}\mathcal{E}_\ell^c}\left|\sum_{\ell=1}^L \bar{\Delta}_L^\ell\right| > 3K_0\sqrt{\frac{d^3\log^3 n}{nL}} + \pi C_5 L n^{-c_1 d}\right] \leq C_5 L n^{-c_1 d} + 2n^{-(9/8)d}.$$

Applying this result and (D.26) to (D.25) via a union bound, we obtain

$$\mathbb{P}\left[\left\{\left|\sum_{\ell=1}^L \Delta_L^\ell\right| > 3K_0\sqrt{\frac{d^3\log^3 n}{nL}} + \pi C_5 L n^{-c_1 d} + \frac{256 C_1}{c_0}\frac{(1+\log L)\log n}{n}\right\}\cap\bigcap_{\ell\in[L-1]}\mathcal{E}_\ell\right]$$
$$\leq C_5 L n^{-c_1 d} + 2n^{-(9/8)d}.$$

If $d \geq 2/c_1$ and $n \geq L_{\max}$, we have $C_5 L n^{-c_1 d} \leq C_5 n^{-c_1 d/2}$; under these condition on $d$ and $n$, we have $\pi C_5 L n^{-c_1 d} \leq \pi c_5 n^{-1}$, and so $\pi C_5 n^{-c_1 d/2} + (256 C_1/c_0)(1+\log L)(\log n)/n \leq C(1+\log L)(\log n)/n$; and if $d \geq 1$ and $n \geq L_{\max}$, we have

$$3K_0\sqrt{\frac{d^3\log^3 n}{nL}} \geq C\frac{(1+\log L)\log n}{n}$$

provided $n \geq C'(C/3K_0)^2 L_{\max}\log L_{\max}$. Under these conditions, our previous bound simplifies to

$$\mathbb{P}\left[\left\{\left|\sum_{\ell=1}^L \Delta_L^\ell\right| > 6K_0\sqrt{\frac{d^3\log^3 n}{nL}}\right\}\cap\bigcap_{\ell\in[L-1]}\mathcal{E}_\ell\right] \leq (2+C_5)n^{-\min\{c_1/2, 9/8\}d}.$$

In particular, applying this bound to (D.22), we have shown that for any $L \geq 2$

$$\mathbb{P}\left[\bigcup_{\ell \in [L]} \mathcal{E}_\ell\right] = \mathbb{P}\left[\bigcup_{\ell \in [L-1]} \mathcal{E}_\ell\right] + (2 + C_5)n^{-\min\{c_1/2, 9/8\}d}.$$

Unraveling the recursion with (D.23) (and worst-casing the constants there), we conclude

$$\mathbb{P}\left[\bigcup_{\ell \in [L]} \mathcal{E}_\ell\right] \leq (2 + C_5)Ln^{-\min\{c_1/2, 9/8\}d},$$

which proves the claim, after possibly choosing $n$ to be larger than another absolute constant multiple of $L_{\max}$ to remove the leading $L$ factor. $\qquad\square$

### D.2.2 BACKWARD FEATURE CONTROL

Having established concentration of the feature norms and the angles between them, it remains to control the inner products of backward features that appear in $\Theta$. The core of the technical approach will once again be martingale concentration. We establish the following control on the backward feature inner products:

**Lemma D.4.** *Fix $\boldsymbol{x}, \boldsymbol{x}' \in \mathbb{S}^{n_0 - 1}$ and denote $\nu = \angle(\boldsymbol{x}, \boldsymbol{x}')$. If $n \geq \max\{KL \log n, K'Ld_b, K''\}$, $d_b \geq K''' \log L$ for suitably chosen $K, K', K'', K'''$ then*

$$\mathbb{P}\left[\bigcap_{\ell=0}^{L-1} \left\{\left\|\boldsymbol{\beta}^\ell(\boldsymbol{x})\right\|_2^2 \leq Cn\right\}\right] \geq 1 - e^{-c\frac{n}{L}}.$$

*If additionally $n, L, d$ satisfy the requirements of lemmas D.3 and E.16, we have*

$$\mathbb{P}\left[\bigcap_{\ell=0}^{L-1} \left\{\left|\left\langle\boldsymbol{\beta}^\ell(\boldsymbol{x}), \boldsymbol{\beta}^\ell(\boldsymbol{x}')\right\rangle - \frac{n}{2}\prod_{i=\ell}^{L-1}\left(1 - \frac{\varphi^{(i)}(\nu)}{\pi}\right)\right| \leq \log^2(n)\sqrt{d^4 Ln}\right\}\right] \geq 1 - e^{-cd}$$

*where $\varphi^{(i)}$ denotes $i$ applications of the angle evolution function defined in lemma E.2, and $c > 0, C$ are absolute constants.*

*Proof.* For $\ell \in [L]$, write $\mathcal{F}^\ell$ for the $\sigma$-algebra generated by all the weights up to layer $\ell$ in the network, i.e., $\boldsymbol{W}^1, \ldots, \boldsymbol{W}^\ell$, with $\mathcal{F}^0$ given by the trivial $\sigma$-algebra. Consider some $\left\langle\beta^{\ell'}(\boldsymbol{x}), \beta^{\ell'}(\boldsymbol{x}')\right\rangle$ for $0 \leq \ell' \leq L - 1$. Defining

$$\boldsymbol{\Gamma}^{\ell:\ell'}(\boldsymbol{x}) = \boldsymbol{P}_{I_\ell(\boldsymbol{x})}\boldsymbol{W}^\ell \boldsymbol{P}_{I_{\ell-1}(\boldsymbol{x})}\ldots \boldsymbol{P}_{I_{\ell'}(\boldsymbol{x})}\boldsymbol{W}^{\ell'},$$

$$\boldsymbol{B}_{\boldsymbol{x}\boldsymbol{x}'}^{\ell:\ell'} = \boldsymbol{\Gamma}^{\ell:\ell'+2}(\boldsymbol{x})\boldsymbol{P}_{I_{\ell'+1}(\boldsymbol{x})}\boldsymbol{P}_{I_{\ell'+1}(\boldsymbol{x}')}\boldsymbol{\Gamma}^{\ell:\ell'+2*}(\boldsymbol{x}'),$$

for $\ell \in \{\ell' + 1, \ldots, L\}$, and setting $\boldsymbol{\Gamma}^{\ell'+1:\ell'+2}(\boldsymbol{x}) = \boldsymbol{I}, \boldsymbol{B}_{\boldsymbol{x}\boldsymbol{x}'}^{\ell':\ell'} = \frac{1}{2}\boldsymbol{I}$, we define the event

$$\tilde{\mathcal{E}}_B^{L+1:\ell'} = \left\{\left\|\boldsymbol{B}_{\boldsymbol{x}\boldsymbol{x}'}^{L:\ell'}\right\|_F^2 \leq C^2 nL\right\} \cap \left\{\left\|\boldsymbol{B}_{\boldsymbol{x}\boldsymbol{x}'}^{L:\ell'}\right\| \leq CL\right\} \cap \left\{\mathrm{tr}\left[\boldsymbol{B}_{\boldsymbol{x}\boldsymbol{x}'}^{L:\ell'}\right] \leq Cn\right\}.$$

Since $\left\langle\beta^{\ell'}(\boldsymbol{x}), \beta^{\ell'}(\boldsymbol{x}')\right\rangle = \boldsymbol{W}^{L+1}\boldsymbol{B}_{\boldsymbol{x}\boldsymbol{x}'}^{L:\ell'}\boldsymbol{W}^{L+1*}$ is a Gaussian chaos in terms of the $\boldsymbol{W}^{L+1}$ variables (and recalling $\boldsymbol{W}^{L+1} \underset{\mathrm{iid}}{\sim} \mathcal{N}(0, \boldsymbol{I})$) and $\tilde{\mathcal{E}}_B^L$ is $\mathcal{F}^L$-measurable, the Hanson-Wright inequality (lemma G.4) gives

$$\mathbb{P}\left[\mathbb{1}_{\tilde{\mathcal{E}}_B^{L+1:\ell'}}\left|\left\langle\beta^{\ell'}(\boldsymbol{x}), \beta^{\ell'}(\boldsymbol{x}')\right\rangle - \mathrm{tr}\left[\boldsymbol{B}_{\boldsymbol{x}\boldsymbol{x}'}^{L:\ell'}\right]\right| \geq C\sqrt{tnL}\right]$$

$$\leq 2\exp\left(-c\min\left\{t, \sqrt{\frac{tn}{L}}\right\}\right) \leq 2e^{-ct}.$$

Using lemma D.28 to bound $\mathbb{P}\left[\left(\tilde{\mathcal{E}}_B^{L+1:\ell'}\right)^c\right]$ from above gives

$$\mathbb{P}\left[\left|\left\langle \boldsymbol{\beta}^{\ell'}(\boldsymbol{x}), \boldsymbol{\beta}^{\ell'}(\boldsymbol{x}')\right\rangle - \mathrm{tr}\left[\boldsymbol{B}_{\boldsymbol{xx}'}^{L:\ell'}\right]\right| > C\sqrt{tnL}\right]$$

$$\leq \mathbb{P}\left[\mathbb{1}_{\tilde{\mathcal{E}}_B^{L+1:\ell'}}\left|\left\langle \boldsymbol{\beta}^{\ell'}(\boldsymbol{x}), \boldsymbol{\beta}^{\ell'}(\boldsymbol{x}')\right\rangle - \mathrm{tr}\left[\boldsymbol{B}_{\boldsymbol{xx}'}^{L:\ell'}\right]\right| \geq C\sqrt{tnL}\right]$$

$$+ \mathbb{P}\left[\mathbb{1}_{\left(\tilde{\mathcal{E}}_B^{L+1:\ell'}\right)^c}\left|\left\langle \boldsymbol{\beta}^{\ell'}(\boldsymbol{x}), \boldsymbol{\beta}^{\ell'}(\boldsymbol{x}')\right\rangle - \mathrm{tr}\left[\boldsymbol{B}_{\boldsymbol{xx}'}^{L:\ell'}\right]\right| \geq C\sqrt{tnL}\right] \quad \text{(D.31)}$$

$$\leq \mathbb{P}\left[\mathbb{1}_{\tilde{\mathcal{E}}_B^{L+1:\ell'}}\left|\left\langle \boldsymbol{\beta}^{\ell'}(\boldsymbol{x}), \boldsymbol{\beta}^{\ell'}(\boldsymbol{x}')\right\rangle - \mathrm{tr}\left[\boldsymbol{B}_{\boldsymbol{xx}'}^{L:\ell'}\right]\right| \geq C\sqrt{tnL}\right] + \mathbb{P}\left[\left(\tilde{\mathcal{E}}_B^{L+1:\ell'}\right)^c\right]$$

$$\leq 2e^{-ct} + Cn^{-c'\frac{n}{L}} \leq C'e^{-c''t}$$

for appropriately chosen constant, if $t \lesssim n\log n/L$. Choosing $t = n/L$ in the bound above and using the bound on $\mathrm{tr}\left[\boldsymbol{B}_{\boldsymbol{xx}}^{L:\ell'}\right]$ from lemma D.28 we obtain

$$\mathbb{P}\left[\left\|\boldsymbol{\beta}^{\ell'}(\boldsymbol{x})\right\|_2^2 \geq 2Cn\right] \leq \mathbb{P}\left[\left\|\boldsymbol{\beta}^{\ell'}(\boldsymbol{x})\right\|_2^2 - \mathrm{tr}\left[\boldsymbol{B}_{\boldsymbol{xx}}^L\right] > Cn\right] + \mathbb{P}\left[\mathrm{tr}\left[\boldsymbol{B}_{\boldsymbol{xx}}^L\right] > Cn\right]$$

$$\leq Ce^{-c'\frac{n}{L}} + C'nL^2e^{-c''\frac{n}{L}} \leq C''nL^2e^{-c''\frac{n}{L}}$$

for appropriate constants. Taking a union bound over the possible values of $\ell'$ proves the first part of the lemma.

We next control $\left|\mathrm{tr}\left[\boldsymbol{B}_{\boldsymbol{xx}'}^{L:\ell'}\right] - \frac{n}{2}\prod_{\ell=\ell'}^{L-1}\left(1 - \frac{\varphi^{(\ell)}(\nu)}{\pi}\right)\right|$ using martingale concentration (in a similar manner to the control of the angles established in previous sections). We write

$$\mathrm{tr}\left[\boldsymbol{B}_{\boldsymbol{xx}'}^{L:\ell'}\right] - \frac{n}{2}\prod_{i=\ell'}^{L-1}\left(1 - \frac{\varphi^{(i)}(\nu)}{\pi}\right) \quad \text{(D.32)}$$

$$= \sum_{\ell=\ell'}^{L-1}\prod_{i=\ell+1}^{L-1}\left(1 - \frac{\varphi^{(i)}(\nu)}{\pi}\right)\left(\mathrm{tr}\left[\boldsymbol{B}_{\boldsymbol{xx}'}^{\ell+1:\ell'}\right] - \left(1 - \frac{\varphi^{(\ell)}(\nu)}{\pi}\right)\mathrm{tr}\left[\boldsymbol{B}_{\boldsymbol{xx}'}^{\ell:\ell'}\right]\right) \equiv \sum_{\ell=\ell'+1}^{L}\Delta_\ell \quad \text{(D.33)}$$

(note the change in the indexing). Consider the filtration $\mathcal{F}^0 \subset \cdots \subset \mathcal{F}^L$ and adapted sequence

$$\overline{\Delta}_\ell = \Delta_\ell - \mathbb{E}\left[\Delta_\ell | \mathcal{F}^{\ell-1}\right], \quad \text{(D.34)}$$

so that

$$\sum_{\ell=\ell'+1}^{L}\Delta_\ell = \sum_{\ell=\ell'+1}^{L}\overline{\Delta}_\ell + \sum_{\ell=\ell'+1}^{L}\mathbb{E}\left[\Delta_\ell | \mathcal{F}^{\ell-1}\right]. \quad \text{(D.35)}$$

We begin by considering the first term in the sum since it takes a distinct form. Denoting by $\boldsymbol{W}_{(:,i)}^{\ell'+1}$ the $i$-th column of $\boldsymbol{W}^{\ell'+1}$, rotational invariance of the Gaussian distribution gives

$$\mathrm{tr}\left[\boldsymbol{B}_{\boldsymbol{xx}'}^{\ell'+1:\ell'}\right] = \mathrm{tr}\left[\boldsymbol{P}_{I_{\ell'+1}(\boldsymbol{x})}\boldsymbol{P}_{I_{\ell'+1}(\boldsymbol{x}')}\right]$$

$$= \mathrm{tr}\left[\boldsymbol{P}_{\boldsymbol{W}^{\ell'+1}\boldsymbol{\alpha}^{\ell'}(\boldsymbol{x})>\boldsymbol{0}}\boldsymbol{P}_{\boldsymbol{W}^{\ell'+1}\boldsymbol{\alpha}^{\ell'}(\boldsymbol{x}')>\boldsymbol{0}}\right]$$

$$\stackrel{d}{=} \mathrm{tr}\left[\boldsymbol{P}_{\boldsymbol{W}_{(:,1)}^{\ell'+1}>\boldsymbol{0}}\boldsymbol{P}_{\boldsymbol{W}_{(:,1)}^{\ell'+1}\cos\nu^{\ell'}+\boldsymbol{W}_{(:,2)}^{\ell'+1}\sin\nu^{\ell'}>\boldsymbol{0}}\right]$$

and hence

$$\mathbb{E}\left[\mathrm{tr}\left[\boldsymbol{B}_{\boldsymbol{xx}'}^{\ell'+1:\ell'}\right]\Big|\mathcal{F}^{\ell'}\right] = \underset{\boldsymbol{W}^{\ell'+1}}{\mathbb{E}}\mathrm{tr}\left[\boldsymbol{B}_{\boldsymbol{xx}'}^{\ell'+1:\ell'}\right] = n\underset{g_1,g_2}{\mathbb{E}}\mathbb{1}_{g_1>0}\mathbb{1}_{g_1\cos\nu^{\ell'}+g_1\sin\nu^{\ell'}>0}$$

where $(g_1, g_2) \sim \mathcal{N}(0, \boldsymbol{I})$. Moving to spherical coordinates, we obtain

$$\underset{\boldsymbol{W}^{\ell'+1}}{\mathbb{E}}\mathrm{tr}\left[\boldsymbol{B}_{\boldsymbol{xx}'}^{\ell'+1:\ell'}\right] = \frac{n}{2\pi}\int_0^\infty\int_{-\frac{\pi}{2}+\nu^{\ell'}}^{\pi/2}e^{-r^2/2}rdrd\theta = \frac{n}{2}\left(1 - \frac{\nu^{\ell'}}{\pi}\right).$$

We now note that conditioned on $\mathcal{F}^{\ell'}$, $\mathrm{tr}\left[\boldsymbol{P}_{\boldsymbol{W}^0_{(:,1)}>\boldsymbol{0}}\boldsymbol{P}_{\boldsymbol{W}^0_{(:,1)}\cos\nu+\boldsymbol{W}^0_{(:,2)}\sin\nu>\boldsymbol{0}}\right]$ is a sum of $n$ independent variables taking values in $\{0,1\}$. An application of Bernstein's inequality for bounded random variables (lemma G.3) then gives

$$\mathbb{P}\left[\left|\overline{\Delta}_{\ell'+1}\right|>\sqrt{nd}\right] \quad =\mathbb{P}\left[\prod_{i=\ell'+1}^{L-1}\left(1-\frac{\varphi^{(i)}(\nu)}{\pi}\right)\left|\mathrm{tr}\left[\boldsymbol{B}^{\ell'+1:\ell'}_{\boldsymbol{x}\boldsymbol{x}'}\right]-\frac{n}{2}\left(1-\frac{\nu^{\ell'}}{\pi}\right)\right|>\sqrt{nd}\right]$$

$$\leq 2\exp\left(-c\frac{nd}{n+\sqrt{nd}}\right)\leq 2e^{-c'd}$$

$$(D.36)$$

for some $c'$, where we used the fact that the angle evolution function $\varphi$ is bounded by $\pi/2$. Note also from Lemma D.3 that

$$\mathbb{P}\left[\left|\mathbb{E}\left[\Delta_{\ell'+1}|\mathcal{F}^{\ell'}\right]-\frac{n}{2}\prod_{i=\ell'}^{L-1}\left(1-\frac{\varphi^{(i)}(\nu)}{\pi}\right)\right|>\sqrt{\frac{d^3\log^3(n)n}{L}}\right]$$

$$=\mathbb{P}\left[\frac{n}{2}\prod_{i=\ell'+1}^{L-1}\left(1-\frac{\varphi^{(i)}(\nu)}{\pi}\right)\left|\frac{\nu^{\ell'}}{\pi}-\frac{\varphi^{(\ell')}(\nu)}{\pi}\right|>\sqrt{\frac{d^3\log^3(n)n}{L}}\right]$$

$$(D.37)$$

$$\leq e^{-cd}$$

for some constant $c$, where we assumed $d>K\log n$ for some $K$.

Having controlled the first term in (D.35), we now proceed to bound the remaining terms. We define events

$$\mathcal{E}^{\ell:\ell'}_B = \begin{array}{c} \left\{\left\|\boldsymbol{\alpha}^{\ell-1}(\boldsymbol{x})\right\|_2\left\|\boldsymbol{\alpha}^{\ell-1}(\boldsymbol{x}')\right\|_2>0\right\} \quad \cap \quad \left\{\left|\varphi^{(\ell-1)}(\nu)-\nu^{\ell-1}\right|\leq C\sqrt{\frac{d^3\log^3 n}{n\ell}}\right\} \\ \cap \quad \left\{\mathrm{tr}\left[\boldsymbol{B}^{\ell-1:\ell'}_{\boldsymbol{x}\boldsymbol{x}'}\right]\leq Cn\right\} \quad \cap \quad \left\{\left\|\boldsymbol{B}^{\ell-1:\ell'}_{\boldsymbol{x}\boldsymbol{x}'}\right\|^2_F\leq C^2n\ell\right\} \\ \cap \quad \left\{\left\|\boldsymbol{B}^{\ell-1:\ell'}_{\boldsymbol{x}\boldsymbol{x}'}\right\|\leq C\ell\right\} \end{array},$$

(which from lemma D.28 hold with high probability). Note that as a consequence of the first event in $\mathcal{E}^{\ell:\ell'}_B$ the angle $\nu^\ell$ is well-defined. Note that $\mathcal{E}^{\ell:\ell'}_B$ is $\mathcal{F}^{\ell-1}$-measurable.

We will first control (D.35) by considering each summand truncated on the respective event $\mathcal{E}^{\ell:\ell'}_B$. Our task is therefore to control

$$\sum_{\ell=\ell'+2}^{L}\mathbb{1}_{\mathcal{E}^{\ell:\ell'}_B}\overline{\Delta}_\ell+\sum_{\ell=\ell'+2}^{L}\mathbb{1}_{\mathcal{E}^{\ell:\ell'}_B}\mathbb{E}\Delta_\ell|\mathcal{F}^{\ell-1}.$$

Since

$$\mathbb{E}\left[\mathbb{1}_{\mathcal{E}^{\ell:\ell'}_B}\overline{\Delta}_\ell\right]=\mathbb{E}\left[\mathbb{E}\left[\mathbb{1}_{\mathcal{E}^{\ell:\ell'}_B}\overline{\Delta}_\ell\Big|\mathcal{F}^{\ell-1}\right]\right]=\mathbb{E}\left[\mathbb{1}_{\mathcal{E}^{\ell:\ell'}_B}\mathbb{E}\left[\overline{\Delta}_\ell\Big|\mathcal{F}^{\ell-1}\right]\right]=0,$$

the first sum is over a zero-mean adapted sequence and hence a martingale, and can thus be controlled using the Azuma-Hoeffding inequality. We will first show that the remaining term is small. We begin by computing

$$\mathbb{1}_{\mathcal{E}^{\ell:\ell'}_B}\mathbb{E}\mathrm{tr}\left[\boldsymbol{B}^{\ell:\ell'}_{\boldsymbol{x}\boldsymbol{x}'}\right]|\mathcal{F}^{\ell-1}=\mathop{\mathbb{E}}_{\boldsymbol{W}^\ell}\mathrm{tr}\left[\mathbb{1}_{\mathcal{E}^{\ell:\ell'}_B}\boldsymbol{B}^{\ell-1:\ell'}_{\boldsymbol{x}\boldsymbol{x}'}\boldsymbol{W}^{\ell*}\boldsymbol{P}_{\boldsymbol{W}^\ell\boldsymbol{\alpha}^{\ell-1}(\boldsymbol{x}')>\boldsymbol{0}}\boldsymbol{P}_{\boldsymbol{W}^\ell\boldsymbol{\alpha}^{\ell-1}(\boldsymbol{x})>\boldsymbol{0}}\boldsymbol{W}^\ell\right]$$

where we used the fact that $\mathcal{E}^{\ell:\ell'}_B\in\mathcal{F}^{\ell-1}$ and is thus independent of $\boldsymbol{W}^\ell$.

There exists a matrix $\boldsymbol{R}$ such that

$$\boldsymbol{R}\boldsymbol{\alpha}^{\ell-1}(\boldsymbol{x})=\left\|\boldsymbol{\alpha}^{\ell-1}(\boldsymbol{x})\right\|_2\widehat{\boldsymbol{e}}_1,\boldsymbol{R}\boldsymbol{\alpha}^{\ell-1}(\boldsymbol{x}')=\left\|\boldsymbol{\alpha}^{\ell-1}(\boldsymbol{x}')\right\|_2\left(\widehat{\boldsymbol{e}}_1\cos\nu^{\ell-1}+\widehat{\boldsymbol{e}}_2\sin\nu^{\ell-1}\right).$$

Rotational invariance of the Gaussian distribution gives

$$\boldsymbol{W}^\ell\boldsymbol{\alpha}^{\ell-1}(\boldsymbol{x})\stackrel{d}{=}\boldsymbol{W}^\ell_{(:,1)}\left\|\boldsymbol{\alpha}^\ell(\boldsymbol{x})\right\|_2,$$

$$\boldsymbol{W}^\ell\boldsymbol{\alpha}^{\ell-1}(\boldsymbol{x}')\stackrel{d}{=}\left\|\boldsymbol{\alpha}^{\ell-1}(\boldsymbol{x}')\right\|_2\left(\boldsymbol{W}^\ell_{(:,1)}\cos\nu^{\ell-1}+\boldsymbol{W}^\ell_{(:,2)}\sin\nu^{\ell-1}\right),$$

where we denote by $\boldsymbol{W}^{\ell}_{(:,i)}$ the $i$-th column of $\boldsymbol{W}^{\ell}$. Defining $\tilde{\boldsymbol{B}}^{\ell-1:\ell'}_{\boldsymbol{x}\boldsymbol{x}'} = \boldsymbol{R}\boldsymbol{B}^{\ell-1:\ell'}_{\boldsymbol{x}\boldsymbol{x}'}\boldsymbol{R}^*$ we have

$$
\begin{aligned}
\mathbb{E}\mathrm{tr}\left[\boldsymbol{B}^{\ell:\ell'}_{\boldsymbol{x}\boldsymbol{x}'}\right]|\mathcal{F}^{\ell-1} \;&=\; \underset{\boldsymbol{W}^{\ell}}{\mathbb{E}}\,\mathrm{tr}\left[\mathbb{1}_{\mathcal{E}^{\ell:\ell'}_{B}}\tilde{\boldsymbol{B}}^{\ell-1:\ell'}_{\boldsymbol{x}\boldsymbol{x}'}\boldsymbol{W}^{\ell*}\boldsymbol{P}_{\boldsymbol{W}^{\ell}_{(:,1)}>\mathbf{0}}\boldsymbol{P}_{\boldsymbol{W}^{\ell}_{(:,1)}\cos\nu^{\ell-1}+\boldsymbol{W}^{\ell}_{(:,2)}\sin\nu^{\ell-1}>\mathbf{0}}\boldsymbol{W}^{\ell}\right] \\
&=\mathbb{1}_{\mathcal{E}^{\ell:\ell'}_{B}}\tilde{\boldsymbol{B}}^{\ell-1:\ell':\ell'}_{\boldsymbol{x}\boldsymbol{x}'ji}\underset{\boldsymbol{W}^{\ell}}{\mathbb{E}}\sum_{ijk}W^{\ell}_{ki}\mathbb{1}_{\boldsymbol{W}^{\ell}_{k1}>0}\mathbb{1}_{\boldsymbol{W}^{\ell}_{k1}\cos\nu^{\ell-1}+\boldsymbol{W}^{\ell}_{k2}\sin\nu^{\ell-1}>0}W^{\ell}_{kj} \\
&=\mathbb{1}_{\mathcal{E}^{\ell:\ell'}_{B}}\sum_{i,j=1}^{n}\tilde{\boldsymbol{B}}^{\ell-1:\ell':\ell'}_{\boldsymbol{x}\boldsymbol{x}'ji}\underset{\boldsymbol{W}^{\ell}}{\mathbb{E}}nW^{\ell}_{1i}\mathbb{1}_{\boldsymbol{W}^{\ell}_{11}>0}\mathbb{1}_{\boldsymbol{W}^{\ell}_{11}\cos\nu^{\ell-1}+\boldsymbol{W}^{\ell}_{12}\sin\nu^{\ell-1}>0}W^{\ell}_{1j} \\
&\doteq\sum_{i,j=1}^{n}\mathbb{1}_{\mathcal{E}^{\ell:\ell'}_{B}}\tilde{\boldsymbol{B}}^{\ell-1:\ell':\ell'}_{\boldsymbol{x}\boldsymbol{x}'ji}Q^{\ell-1}_{ij}.
\end{aligned}
$$

If $i\notin\{1,2\}$ we get (with the square brackets denoting indicators)

$$
Q^{\ell-1}_{ij} = \underset{\boldsymbol{W}^{\ell}}{\mathbb{E}}2\delta_{ij}\left[\boldsymbol{W}^{\ell}_{11}>0\right]\left[\boldsymbol{W}^{\ell}_{11}\cos\nu^{\ell-1}+\boldsymbol{W}^{\ell}_{12}\sin\nu^{\ell-1}>0\right] = \delta_{ij}\left(1-\frac{\nu^{\ell-1}}{\pi}\right).
$$

If $i\in\{1,2\}$ then the $Q^{\ell-1}_{ij}\neq 0$ only if $j\in\{1,2\}$. In these cases we have

$$
\begin{aligned}
Q^{\ell-1}_{11} \;&=\; \underset{\boldsymbol{W}^{\ell}}{\mathbb{E}}n\left(W^{\ell}_{11}\right)^2\left[\boldsymbol{W}^{\ell}_{11}>0\right]\left[\boldsymbol{W}^{\ell}_{11}\cos\nu^{\ell-1}+\boldsymbol{W}^{\ell}_{12}\sin\nu^{\ell-1}>0\right] \\
&=2\underset{\boldsymbol{W}^{\ell}}{\mathbb{E}}g_1^2\left[g_1>0\right]\left[g_1\cos\nu^{\ell-1}+g_2\sin\nu^{\ell-1}>0\right]
\end{aligned}
$$

where $(g_1,g_2)\sim\mathcal{N}(0,\boldsymbol{I})$. Moving to spherical coordinates, we obtain

$$
Q^{\ell-1}_{11} = \frac{1}{\pi}\int_{0}^{\infty}\int_{-\frac{\pi}{2}+\nu^{\ell-1}}^{\pi/2}e^{-r^2/2}r^3\cos^2\theta\,drd\theta = \frac{\pi-\nu^{\ell-1}+\sin\nu^{\ell-1}\cos\nu^{\ell-1}}{\pi},
$$

and similarly

$$
Q^{\ell-1}_{22} = \frac{1}{\pi}\int_{0}^{\infty}\int_{-\frac{\pi}{2}+\nu}^{\pi/2}e^{-r^2/2}r^3\sin^2\theta\,drd\theta = \frac{\pi-\nu^{\ell-1}-\sin\nu^{\ell-1}\cos\nu^{\ell-1}}{\pi}
$$

$$
Q^{\ell-1}_{12} = Q^{\ell-1}_{21} = \underset{\boldsymbol{W}^{\ell}}{\mathbb{E}}nW^{\ell}_{11}\left[\boldsymbol{W}^{\ell}_{11}>0\right]\left[\boldsymbol{W}^{\ell}_{11}\cos\nu^{\ell-1}+\boldsymbol{W}^{\ell}_{12}\sin\nu^{\ell-1}>0\right]\boldsymbol{W}^{\ell}_{12}
$$

$$
= \frac{1}{\pi}\int_{0}^{\infty}\int_{-\frac{\pi}{2}+\nu^{\ell-1}}^{\pi/2}e^{-r^2/2}r^3\sin\theta\cos\theta\,drd\theta = \frac{1}{2\pi}\int_{-\frac{\pi}{2}+\nu^{\ell-1}}^{\pi/2}\sin\theta\cos\theta\,d\theta = \frac{\sin^2\nu^{\ell-1}}{2\pi}.
$$

Combining terms and using $\mathrm{tr}\left[\boldsymbol{B}^{\ell-1:\ell'}_{\boldsymbol{x}\boldsymbol{x}'}\right] = \mathrm{tr}\left[\tilde{\boldsymbol{B}}^{\ell-1:\ell'}_{\boldsymbol{x}\boldsymbol{x}'}\right]$ we obtain

$$
\mathbb{1}_{\mathcal{E}^{\ell:\ell'}_{B}}\mathbb{E}\left[\mathrm{tr}\left[\boldsymbol{B}^{\ell:\ell'}_{\boldsymbol{x}\boldsymbol{x}'}\right]|\mathcal{F}^{\ell-1}\right] = \mathbb{1}_{\mathcal{E}^{\ell:\ell'}_{B}}\left(\begin{array}{c}\dfrac{\pi-\nu^{\ell-1}}{\pi}\mathrm{tr}\left[\boldsymbol{B}^{\ell-1:\ell'}_{\boldsymbol{x}\boldsymbol{x}'}\right] \\ +\dfrac{\sin\nu^{\ell-1}\cos\nu^{\ell-1}}{\pi}\left(\tilde{B}^{\ell-1:\ell'}_{\boldsymbol{x}\boldsymbol{x}'11}-\tilde{B}^{\ell-1:\ell'}_{\boldsymbol{x}\boldsymbol{x}'22}\right) \\ +\dfrac{\sin^2\nu^{\ell-1}}{2\pi}\left(\tilde{B}^{\ell-1:\ell'}_{\boldsymbol{x}\boldsymbol{x}'12}+\tilde{B}^{\ell-1:\ell'}_{\boldsymbol{x}\boldsymbol{x}'21}\right)\end{array}\right),
$$

hence

$$
\mathbb{1}_{\mathcal{E}^{\ell:\ell'}_{B}}\underset{\boldsymbol{W}^{\ell}}{\mathbb{E}}\left[\mathrm{tr}\left[\boldsymbol{B}^{\ell:\ell'}_{\boldsymbol{x}\boldsymbol{x}'}\right]-\left(1-\frac{\varphi^{\ell-1}(\nu)}{\pi}\right)\mathrm{tr}\left[\boldsymbol{B}^{\ell-1:\ell'}_{\boldsymbol{x}\boldsymbol{x}'}\right]\right]
$$

$$
=\mathbb{1}_{\mathcal{E}^{\ell:\ell'}_{B}}\left[\begin{array}{c}\frac{\varphi^{\ell-1}(\nu)-\nu^{\ell-1}}{\pi}\mathrm{tr}\left[\boldsymbol{B}^{\ell-1:\ell'}_{\boldsymbol{x}\boldsymbol{x}'}\right] \\ +\frac{\sin\nu^{\ell-1}\cos\nu^{\ell-1}}{\pi}\left(\tilde{B}^{\ell-1:\ell'}_{\boldsymbol{x}\boldsymbol{x}'11}-\tilde{B}^{\ell-1:\ell'}_{\boldsymbol{x}\boldsymbol{x}'22}\right) \\ +\frac{\sin^2\nu^{\ell-1}}{2\pi}\left(\tilde{B}^{\ell-1:\ell'}_{\boldsymbol{x}\boldsymbol{x}'12}+\tilde{B}^{\ell-1:\ell'}_{\boldsymbol{x}\boldsymbol{x}'21}\right)\end{array}\right]
$$

On $\mathcal{E}_B^{\ell:\ell'}$, the bound on $\left|\varphi^{\ell-1}(\nu) - \nu^{\ell-1}\right|$ and lemma C.12 give $\nu^{\ell-1} \leq \frac{C}{\ell}$ a.s.. Additionally, on this event $\max_{i,j\in[n]} \left|\tilde{B}_{\boldsymbol{xx'}ij}^{\ell-1:\ell'}\right| \leq \left\|\boldsymbol{B}_{\boldsymbol{xx'}}^{\ell-1:\ell'}\right\| \leq C\ell$ a.s.. It follows that

$$\left|\mathbb{1}_{\mathcal{E}_B^{\ell:\ell'}} \underset{\boldsymbol{W}^\ell}{\mathbb{E}}\left[\operatorname{tr}\left[\boldsymbol{B}_{\boldsymbol{xx'}}^{\ell:\ell'}\right] - \left(1 - \frac{\varphi^{\ell-1}(\nu)}{\pi}\right)\operatorname{tr}\left[\boldsymbol{B}_{\boldsymbol{xx'}}^{\ell-1:\ell'}\right]\right]\right| \tag{D.38}$$

$$\leq C^2\sqrt{\frac{d^3 n \log^3 n}{\ell}} + \frac{2C^2}{\pi} + \frac{C^3}{\pi\ell} \leq C'\sqrt{\frac{d^3 n \log^3 n}{\ell}} \tag{D.39}$$

almost surely, and hence restoring a constant factor with magnitude bounded by 1, we have

$$\mathbb{1}_{\mathcal{E}^{\ell:\ell'}}\left|\mathbb{E}\Delta_\ell|\mathcal{F}^{\ell-1}\right| \tag{D.40}$$

$$= \prod_{i=\ell}^{L-1}\left(1 - \frac{\varphi^{(i)}(\nu)}{\pi}\right)\left|\mathbb{1}_{\mathcal{E}_B^{\ell:\ell'}} \underset{\boldsymbol{W}^\ell}{\mathbb{E}}\left[\operatorname{tr}\left[\boldsymbol{B}_{\boldsymbol{xx'}}^{\ell:\ell'}\right] - \left(1 - \frac{\varphi^{\ell-1}(\nu)}{\pi}\right)\operatorname{tr}\left[\boldsymbol{B}_{\boldsymbol{xx'}}^{\ell-1:\ell'}\right]\right]\right| \tag{D.41}$$

$$\underset{a.s.}{\leq} C'\sqrt{\frac{d^3 n \log^3 n}{\ell}}. \tag{D.42}$$

Using lemma D.28 to bound $\mathbb{P}\left[\left(\mathcal{E}_B^{\ell:\ell'}\right)^c\right]$ from above then gives

$$\mathbb{P}\left[\left|\mathbb{E}\Delta_\ell|\mathcal{F}^{\ell-1}\right| > C'\sqrt{\frac{d^3 n \log^3 n}{\ell}}\right] < \mathbb{P}\left[\left(\mathcal{E}_B^{\ell:\ell'}\right)^c\right] \leq C'n^{-cd}. \tag{D.43}$$

An application of the triangle inequality and union bound then give

$$\begin{aligned}
\mathbb{P}\left[\left|\sum_{\ell=\ell'+2}^{L}\mathbb{E}\Delta_\ell|\mathcal{F}^{\ell-1}\right| > C'\sqrt{d^3 L n \log^3 n}\right] \quad &\leq \mathbb{P}\left[\sum_{\ell=\ell'+2}^{L}\left|\mathbb{E}\Delta_\ell|\mathcal{F}^{\ell-1}\right| > C'\sqrt{d^3 L n \log^3 n}\right] \\
&\leq \sum_{\ell=\ell'+2}^{L}\mathbb{P}\left[\left|\mathbb{E}\Delta_\ell|\mathcal{F}^{\ell-1}\right| > C'\sqrt{\frac{d^3 n \log^3 n}{L}}\right] \\
&\leq \sum_{\ell=\ell'+2}^{L}\mathbb{P}\left[\left(\mathcal{E}_B^{\ell:\ell'}\right)^c\right] \\
&\leq CLn^{-cd}
\end{aligned} \tag{D.44}$$

for some constants $c, C$.

We proceed to control the remaining terms in (D.35), namely $\sum_{\ell=\ell'+2}^{L}\overline{\Delta}_\ell$. Aiming to apply martingale concentration, we require an almost sure bound on the summands, which we achieve by truncation. Towards this end, we define an event

$$\mathcal{G}_\ell = \left\{|\Delta_\ell| \leq C\sqrt{d\ell} + C'\sqrt{\frac{d^3 n \log^3 n}{\ell}}\right\}.$$

Combining (D.43) and the result of lemma D.29 (after taking an expectation) we have

$$\mathbb{P}[\mathcal{G}_\ell] \geq 1 - \mathbb{P}\left[\left|\mathbb{E}\Delta_\ell|\mathcal{F}^{\ell-1}\right| > C'\sqrt{\frac{d^3 n \log^3 n}{\ell}}\right] - \mathbb{P}\left[|\overline{\Delta}_\ell| > C\sqrt{d\ell}\right]$$

$$\geq 1 - C''n^{-cd} - C'''e^{-c'd} \geq 1 - C''''e^{-c'd} \tag{D.45}$$

for appropriate constants. We now decompose the sum that we would like to bound:

$$\left|\sum_{\ell=\ell'+2}^{L}\overline{\Delta}_\ell\right| \leq \quad \underbrace{\left|\sum_{\ell=\ell'+2}^{L}\Delta_\ell - \Delta_\ell\mathbb{1}_{\mathcal{G}_\ell}\right|}_{\Sigma_1} \\ + \underbrace{\left|\sum_{\ell=\ell'+2}^{L}\Delta_\ell\mathbb{1}_{\mathcal{G}_\ell} - \mathbb{E}\left[\Delta_\ell\mathbb{1}_{\mathcal{G}_\ell}|\mathcal{F}^{\ell-1}\right]\right|}_{\Sigma_2} \\ + \underbrace{\left|\sum_{\ell=\ell'+2}^{L}\mathbb{E}\left[\Delta_\ell\mathbb{1}_{\mathcal{G}_\ell}|\mathcal{F}^{\ell-1}\right] - \mathbb{E}\left[\Delta_\ell|\mathcal{F}^{\ell-1}\right]\right|}_{\Sigma_3} \quad . \tag{D.46}$$

Since each summand in $\Sigma_1$ are equal to zero on the respective truncation event, a union bound and (D.45) give

$$\mathbb{P}\left[\left|\sum_{\ell=\ell'+2}^{L}\Delta_\ell - \Delta_\ell\mathbb{1}_{\mathcal{G}_\ell}\right| > 0\right] \leq \mathbb{P}\left[\bigcup_{\ell=\ell'+2}^{L}\mathcal{G}_\ell^c\right] \leq \sum_{\ell=\ell'+2}^{L}\mathbb{P}\left[\mathcal{G}_\ell^c\right] \leq LCe^{-cd} \tag{D.47}$$

for some constants. The term $\Sigma_2$ is a sum of almost surely bounded martingale differences. We can apply the Azuma-Hoeffding inequality (lemma G.8) directly to conclude

$$\mathbb{P}\left[\left|\sum_{\ell=\ell'+2}^{L}\Delta_\ell\mathbb{1}_{\mathcal{G}_\ell} - \mathbb{E}\Delta_\ell\mathbb{1}_{\mathcal{G}_\ell}|\mathcal{F}^{\ell-1}\right| > d^2\sqrt{nL\log^3 n\log L}\right] \tag{D.48}$$

$$\leq \exp\left(-\frac{d^4 nL\log^3 n\log L}{2\sum_{\ell=\ell'+2}^{L}\left(C\sqrt{d\ell} + C'\sqrt{\frac{d^3 n\log^3 n}{\ell}}\right)^2}\right) \leq e^{-cd}. \tag{D.49}$$

Considering a single summand in $\Sigma_3$, Jensen's inequality and the Cauchy-Schwarz inequality give

$$\left|\mathbb{E}\left[\Delta_\ell\mathbb{1}_{\mathcal{G}_\ell} - \Delta_\ell\middle|\mathcal{F}^{\ell-1}\right]\right| = \left|\mathbb{E}\left[\Delta_\ell\mathbb{1}_{\mathcal{G}_\ell^c}\middle|\mathcal{F}^{\ell-1}\right]\right|$$

$$\underset{a.s.}{\leq} \mathbb{E}\left[|\Delta_\ell|\mathbb{1}_{\mathcal{G}_\ell^c}\middle|\mathcal{F}^{\ell-1}\right] \underset{a.s.}{\leq} \left(\mathbb{E}\left[\mathbb{1}_{\mathcal{G}_\ell^c}\middle|\mathcal{F}^{\ell-1}\right]\right)^{1/2}\left(\mathbb{E}\left[\Delta_\ell^2\middle|\mathcal{F}^{\ell-1}\right]\right)^{1/2}. \tag{D.50}$$

This is an $\mathcal{F}^{\ell-1}$-measurable function, and we can show that it is small on the event $\mathcal{E}_B^{\ell:\ell'} \in \mathcal{F}^{\ell-1}$. To control the first factor, we note that

$$\mathbb{1}_{\mathcal{E}_B^{\ell:\ell'}}\mathbb{E}\left[\mathbb{1}_{\mathcal{G}_\ell^c}\middle|\mathcal{F}^{\ell-1}\right] \quad =\mathbb{E}\left[\mathbb{1}_{\mathcal{G}_\ell^c\cap\mathcal{E}_B^{\ell:\ell'}}\middle|\mathcal{F}^{\ell-1}\right]$$

$$=\mathbb{P}\left[\left\{|\Delta_\ell| > C\sqrt{d\ell} + C'\sqrt{\frac{d^3 n\log^3 n}{\ell}}\right\}\cap\mathcal{E}_B^{\ell:\ell'}\middle|\mathcal{F}^{\ell-1}\right]$$

$$\leq \frac{\mathbb{P}\left[\left\{\left|\mathbb{1}_{\mathcal{E}_B^{\ell:\ell'}}\Delta_\ell\right| > C\sqrt{d\ell} + C'\sqrt{\frac{d^3 n\log^3 n}{\ell}}\right\}\cap\mathcal{E}_B^{\ell:\ell'}\middle|\mathcal{F}^{\ell-1}\right]}{+\mathbb{P}\left[\left\{\left|\mathbb{1}_{(\mathcal{E}_B^{\ell:\ell'})^c}\Delta_\ell\right| > 0\right\}\cap\mathcal{E}_B^{\ell:\ell'}\middle|\mathcal{F}^{\ell-1}\right]}$$

$$\leq\mathbb{P}\left[\left|\mathbb{1}_{\mathcal{E}_B^{\ell:\ell'}}\Delta_\ell\right| > C\sqrt{d\ell} + C'\sqrt{\frac{d^3 n\log^3 n}{\ell}}\middle|\mathcal{F}^{\ell-1}\right] \tag{D.51}$$

$$\leq \frac{\mathbb{P}\left[\left|\mathbb{1}_{\mathcal{E}_B^{\ell:\ell'}}\overline{\Delta}_\ell\right| > C\sqrt{d\ell}\middle|\mathcal{F}^{\ell-1}\right]}{+\mathbb{P}\left[\mathbb{1}_{\mathcal{E}_B^{\ell:\ell'}}\left|\mathbb{E}\Delta_\ell|\mathcal{F}^{\ell-1}\right| > C'\sqrt{\frac{d^3 n\log^3 n}{\ell}}\middle|\mathcal{F}^{\ell-1}\right]}$$

$$\underset{a.s.}{\leq}\mathbb{P}\left[\left|\mathbb{1}_{\mathcal{E}_B^{\ell:\ell'}}\overline{\Delta}_\ell\right| > C\sqrt{d\ell}\right] \underset{a.s.}{\leq} C'e^{-cd}$$

where to obtain the second to last line we used the definition of $\Delta_\ell$, then used Lemma D.29 and (D.40) to bound the first and second term almost surely.

We proceed to control the second factor in (D.50), by bounding

$$\mathbb{1}_{\mathcal{E}_B^{\ell:\ell'}}\mathbb{E}\left[\Delta_\ell^2\middle|\mathcal{F}^{\ell-1}\right]$$

$$=\mathbb{1}_{\mathcal{E}_B^{\ell:\ell'}}\prod_{i=\ell}^{L-1}\left(1-\frac{\varphi^{(i)}(\nu)}{\pi}\right)^2\mathbb{E}_{\boldsymbol{W}^\ell}\left[\left(\mathrm{tr}\left[\boldsymbol{B}_{\boldsymbol{x}\boldsymbol{x}'}^{\ell:\ell'}\right]-\left(1-\frac{\varphi^{(\ell-1)}(\nu)}{\pi}\right)\mathrm{tr}\left[\boldsymbol{B}_{\boldsymbol{x}\boldsymbol{x}'}^{\ell-1:\ell'}\right]\right)^2\right]$$

$$\underset{a.s.}{\leq}\mathbb{1}_{\mathcal{E}_B^{\ell:\ell'}}\mathbb{E}_{\boldsymbol{W}^\ell}\left[\left(\mathrm{tr}\left[\boldsymbol{B}_{\boldsymbol{x}\boldsymbol{x}'}^{\ell:\ell'}\right]-\left(1-\frac{\varphi^{(\ell-1)}(\nu)}{\pi}\right)\mathrm{tr}\left[\boldsymbol{B}_{\boldsymbol{x}\boldsymbol{x}'}^{\ell-1:\ell'}\right]\right)^2\right]$$

$$\underset{a.s.}{\leq}4\,\mathbb{E}_{\boldsymbol{W}^\ell}\left[\begin{array}{c}\left(\mathbb{1}_{\mathcal{E}_B^{\ell:\ell'}}\left[\mathrm{tr}\left[\boldsymbol{B}_{\boldsymbol{x}\boldsymbol{x}'}^{\ell:\ell'}\right]-\mathbb{E}_{\boldsymbol{W}^\ell}\left[\mathrm{tr}\left[\boldsymbol{B}_{\boldsymbol{x}\boldsymbol{x}'}^{\ell:\ell'}\right]\right]\right]\right)^2\\+\left(\mathbb{1}_{\mathcal{E}_B^{\ell:\ell'}}\left[\mathbb{E}_{\boldsymbol{W}^\ell}\left[\mathrm{tr}\left[\boldsymbol{B}_{\boldsymbol{x}\boldsymbol{x}'}^{\ell:\ell'}\right]\right]-\left(1-\frac{\varphi^{(\ell-1)}(\nu)}{\pi}\right)\mathrm{tr}\left[\boldsymbol{B}_{\boldsymbol{x}\boldsymbol{x}'}^{\ell-1:\ell'}\right]\right]\right)^2\end{array}\right].$$

Using (D.38) and (D.148) to bound the integrand above, we have

$$\mathbb{P}\left[\mathbb{1}_{\mathcal{E}_B^{\ell:\ell'}}\mathbb{E}\left[\Delta_\ell^2\middle|\mathcal{F}^{\ell-1}\right]>C\left(d\ell+\frac{d^3n\log^3 n}{\ell}\right)\right]\leq C'e^{-cd}$$

for appropriate constants. Combining (D.51) and the above bound gives

$$\mathbb{P}\left[\mathbb{1}_{\mathcal{E}_B^{\ell:\ell'}}\left|\mathbb{E}\left[\mathbb{1}_{\mathcal{G}_\ell}\Delta_\ell-\Delta_\ell\middle|\mathcal{F}^{\ell-1}\right]\right|>C\sqrt{d\ell+\frac{d^3n\log^3 n}{\ell}}e^{-cd}\right]\leq C'e^{-c'd}$$

for some $c,c',C,C'$, and using lemma D.28

$$\mathbb{P}\left[\left|\mathbb{E}\left[\mathbb{1}_{\mathcal{G}_\ell}\Delta_\ell-\Delta_\ell\middle|\mathcal{F}^{\ell-1}\right]\right|>C\sqrt{d\ell+\frac{d^3n\log^3 n}{\ell}}e^{-cd}\right]\leq C'e^{-c'd}+\mathbb{P}\left[(\mathcal{E}_B^{\ell:\ell'})^c\right]$$

$$\leq C'e^{-c'd}+C''n^{-c''d}\leq C'''e^{-c'''d}$$

for appropriate constants. An application of the triangle inequality and a union bound (and introducing some slack to simplify the expression) then gives

$$\mathbb{P}\left[\left|\sum_{\ell=\ell'+2}^{L}\mathbb{E}\left[\mathbb{1}_{\mathcal{G}_\ell}\Delta_\ell-\Delta_\ell\middle|\mathcal{F}^{\ell-1}\right]\right|>CL\sqrt{d^3n\log^3 n}e^{-cd}\right]\leq C'Le^{-cd}$$

for some constants. Combining this bound with (D.47) and (D.48) gives

$$\mathbb{P}\left[\left|\sum_{\ell=\ell'+2}^{L}\overline{\Delta}_\ell\right|>d^2\sqrt{nL\log^3 n\log L}+CL\sqrt{d^3n\log^3 n}e^{-cd}\right]$$

$$\leq C'Le^{-c'd}+e^{-cd}+C''Le^{-c''d}\leq C'''Le^{-c'''d}$$

$$\mathbb{P}\left[\left|\sum_{\ell=\ell'+2}^{L}\overline{\Delta}_\ell\right|>Cd^2\sqrt{Ln\log^3 n\log L}\right]\leq C'Le^{-cd}.$$

where in the last inequality we assumed $K\log L\leq d$. Combining this with (D.44), we obtain

$$\mathbb{P}\left[\left|\sum_{\ell=\ell'+2}^{L}\Delta_\ell\right|>Cd^2\sqrt{Ln\log^3 n\log L}\right]\leq C''Ln^{-c'd}+C'''Le^{-c''d}\leq C'Le^{-cd}$$

for appropriate constants. This bound all the terms in the sum (D.32) aside from the first one. The first term is bounded in (D.36), (D.37), and the fluctuations due to the last layer are bounded in (D.31). Combining all of these gives

$$\mathbb{P}\left[\left|\left\langle\boldsymbol{\beta}^{\ell'}(\boldsymbol{x}),\boldsymbol{\beta}^{\ell'}(\boldsymbol{x}')\right\rangle-\frac{n}{2}\prod_{i=\ell'}^{L-1}\left(1-\frac{\varphi^{(i)}(\nu)}{\pi}\right)\right|>Cd^2\sqrt{Ln\log^3 n\log L}\right]$$

$$\leq \mathbb{P}\left[\left|\left\langle \boldsymbol{\beta}^{\ell'}(\boldsymbol{x}), \boldsymbol{\beta}^{\ell'}(\boldsymbol{x}')\right\rangle - \operatorname{tr}\left[\boldsymbol{B}_{\boldsymbol{x}\boldsymbol{x}'}^{L-1:\ell'}\right]\right| > \frac{C}{4}d^2\sqrt{Ln\log^3 n\log L}\right]$$

$$+ \mathbb{P}\left[|\Delta_{\ell'+1}| \leq \frac{C}{3}d^2\sqrt{Ln\log^3 n\log L}\right] + \mathbb{P}\left[\left|\sum_{\ell=\ell'+2}^{L}\overline{\Delta}_\ell\right| \leq \frac{C}{4}d^2\sqrt{Ln\log^3 n\log L}\right]$$

$$+ \mathbb{P}\left[\left|\sum_{\ell=\ell'+2}^{L}\mathbb{E}\Delta_\ell|\mathcal{F}^{\ell-1}\right| \leq \frac{C}{4}d^2\sqrt{Ln\log^3 n\log L}\right]$$

$$\leq C'Le^{-cd}$$

after worsening constants. A final union bound over $\ell'$ and assuming $d \geq K\log L$ gives

$$\mathbb{P}\left[\bigcap_{\ell'=0}^{L-1}\left\{\left|\left\langle \boldsymbol{\beta}^{\ell'}(\boldsymbol{x}), \boldsymbol{\beta}^{\ell'}(\boldsymbol{x}')\right\rangle - \frac{n}{2}\prod_{i=\ell'}^{L-1}\left(1 - \frac{\varphi^{(i)}(\nu)}{\pi}\right)\right| \geq Cd^2\sqrt{Ln\log^3 n\log L}\right\}\right] \geq 1 - C'e^{-cd}$$

for appropriately chosen $c, C', K$. If we additionally assume $n > L$ we obtain the desired result. $\quad\square$

## D.3 Uniformization Estimates

### D.3.1 Nets and Covering Numbers

We appeal to Lemma C.4 to obtain estimates for the covering number of $\mathcal{M}$, which we will use throughout this section. In the remainder of this section, we will use the notation $N_\varepsilon$ to denote the $\varepsilon$-nets for $\mathcal{M}$ constructed in Lemma C.4, and for any $\bar{\boldsymbol{x}} \in N_\varepsilon$, we will also use the notation $\mathcal{N}_\varepsilon(\bar{\boldsymbol{x}}) = \mathbb{B}(\bar{\boldsymbol{x}}, \varepsilon) \cap \mathcal{M}_\square$, where $\square \in \{+, -\}$ is the component of $\bar{\boldsymbol{x}}$, to denote the relevant connected neighborhood of the specific point in the net we are considering. Here we are implicitly assuming that $\mathcal{M}_\pm$ are themselves connected, but this construction evidently generalizes to cases where $\mathcal{M}_\pm$ themselves have a positive number of connected components, as treated in Lemma C.4. Focusing on this simpler case in the sequel will allow us to keep our notation concise.

### D.3.2 Controlling Support Changes Uniformly

The quantities we have studied in Appendix D.2 are challenging to uniformize due to discontinuities in the support projections $\boldsymbol{P}_{I_\ell(\,\cdot\,)}$. We will get around this difficulty by carefully tracking (with high probability) how much the supports can change by when we move away from the points in our net $N_\varepsilon$. It seems intuitively obvious that when $\varepsilon$ is exponentially small in all problem parameters, there should be almost no support changes when moving away from our net; the challenge is to show that this property also holds when $\varepsilon$ is not so relatively small.

Introduce the following notation for the network preactivations at level $\ell$, where $\ell \in [L]$:

$$\boldsymbol{\rho}^\ell(\boldsymbol{x}) = \boldsymbol{W}^\ell\boldsymbol{\alpha}^{\ell-1}(\boldsymbol{x}),$$

so that $\boldsymbol{\alpha}^\ell(\boldsymbol{x}) = [\boldsymbol{\rho}^\ell(\boldsymbol{x})]_+$. We also let $\mathcal{F}^\ell$ denote the $\sigma$-algebra generated by all weight matrices up to level $\ell$ in the network, and let $\mathcal{F}^0$ denote the trivial $\sigma$-algebra.

**Definition D.1.** Let $\varepsilon, \Delta > 0$, and let $\bar{\boldsymbol{x}} \in N_\varepsilon$. For $\ell \in [L]$, a feature $(\boldsymbol{\alpha}^\ell(\bar{\boldsymbol{x}}))_i$ is called $\Delta$-*risky* if $|(\boldsymbol{\rho}^\ell(\bar{\boldsymbol{x}}))_i| \leq \Delta$; otherwise, it is called $\Delta$-*stable*. If for all $\boldsymbol{x} \in \mathcal{N}_\varepsilon(\bar{\boldsymbol{x}})$ we have

$$\forall \ell' \in [\ell], \; \left\|\boldsymbol{\rho}^\ell(\boldsymbol{x}) - \boldsymbol{\rho}^\ell(\bar{\boldsymbol{x}})\right\|_\infty \leq \Delta,$$

we say that *stable sign consistency holds up to layer* $\ell$. We abbreviate this condition as $\mathsf{SSC}(\ell, \varepsilon, \Delta)$ at $\bar{\boldsymbol{x}}$, with the dependence on $\bar{\boldsymbol{x}}$, $\varepsilon$, and $\Delta$ suppressed when it is clear from context.

If $\mathsf{SSC}(\ell)$ holds at $\bar{\boldsymbol{x}}$ and if $(\boldsymbol{\alpha}^{\ell'}(\bar{\boldsymbol{x}}))_i$ is stable, we can write for any $\boldsymbol{x} \in \mathcal{N}_\varepsilon(\bar{\boldsymbol{x}})$

$$\operatorname{sign}\left(\left(\boldsymbol{\rho}^{\ell'}(\boldsymbol{x})\right)_i\right) = \operatorname{sign}\left(\left(\boldsymbol{\rho}^{\ell'}(\bar{\boldsymbol{x}})\right)_i + \left(\left(\boldsymbol{\rho}^{\ell'}(\boldsymbol{x})\right)_i - \left(\boldsymbol{\rho}^{\ell'}(\bar{\boldsymbol{x}})\right)_i\right)\right) = \operatorname{sign}\left(\left(\boldsymbol{\rho}^{\ell'}(\bar{\boldsymbol{x}})\right)_i\right),$$

so that no stable feature supports change on $\mathcal{N}_\varepsilon(\bar{\boldsymbol{x}})$, and we only need to consider changes due to the risky features. Moreover, observe that

$$\mathbb{P}\left[(\boldsymbol{\rho}^\ell(\bar{\boldsymbol{x}}))_i \in \{\pm\Delta\}\right] = \mathbb{E}\left[\mathbb{P}\left[\|\boldsymbol{\alpha}^{\ell-1}(\boldsymbol{x})\|_2\langle \boldsymbol{e}_i, \boldsymbol{g}\rangle \in \{\pm\Delta\} \mid \mathcal{F}^{\ell-1}\right]\right] = 0, \tag{D.52}$$

where $g \sim \mathcal{N}(0, (2/n)I)$ is independent of everything else in the problem, since $\Delta > 0$. It follows that when considering the network features over any countable collection of points $\bar{x} \in \mathcal{M}$, we have almost surely that the risky features are witnessed in the interior of $[-\Delta, +\Delta]$.

Below, we will show that with appropriate choices of $\varepsilon$ and $\Delta$, with very high probability: (i) each point in the net $\bar{x}$ has very few risky features; and (ii) $\mathsf{SSC}(L)$ holds uniformly over the net under reasonable conditions involving $n, L, d$. We write $R_\ell(\bar{x}, \Delta) \subset [n]$ for the random variable consisting of the set of indices of $\Delta$-risky features at level $\ell$ with input $\bar{x} \in N_\varepsilon$.

**Lemma D.5.** *There is an absolute constant $K > 0$ such that for any $\bar{x} \in \mathcal{M}$ and any $d > 0$, if $n \geq \max\{KdL, 4\}$ and $\Delta \leq d \log n / (6n^{3/2}L)$, then one has*

$$\mathbb{P}\left[\sum_{\ell=1}^{L} |R_\ell(\bar{x}, \Delta)| > d \log n\right] \leq 2n^{-d} + L^2 e^{-cn/L}.$$

*Proof.* For any $\bar{x} \in N_\varepsilon$, Lemma D.2 (with a suitable choice of $d$ in that context) gives

$$\mathbb{P}\left[\left|\|\alpha^\ell(\bar{x})\|_2 - 1\right| > \frac{1}{2}\right] \leq C\ell e^{-c\frac{n}{\ell}},$$

so that if additionally $n \geq (2/c)\ell \log(C)$, one has

$$\mathbb{P}\left[\left|\|\alpha^\ell(\bar{x})\|_2 - 1\right| > \frac{1}{2}\right] \leq \ell e^{-c\frac{n}{\ell}}. \tag{D.53}$$

Let $\mathcal{G}_\ell = \{1/2 \leq \|\alpha^\ell(\bar{x})\|_2 \leq 2\}$, so that $\mathcal{G}_\ell$ is $\mathcal{F}^\ell$-measurable, and $\mathcal{G} = \cap_{\ell \in [L-1]}\mathcal{G}_\ell$; then by (D.53) and a union bound, we have $\mathbb{P}[\mathcal{G}] \geq 1 - L^2 e^{-cn/L}$. We also let $\mathcal{G}_0 = \varnothing^c$. For $i \in [n]$ and $\ell \in [L]$, consider the random variables $X_{i\ell} = |(\rho^\ell(\bar{x}))_i|$, and moreover define

$$\tilde{X}_{i\ell} = \frac{X_{i\ell}}{\|\alpha^{\ell-1}(\bar{x})\|_2}\mathbb{1}_{\mathcal{G}_{\ell-1}}.$$

We have $\sum_{i,\ell} \mathbb{1}_{X_{i\ell} \leq \Delta} = \sum_\ell |R_\ell(\bar{x})|$, which is the total number of $\Delta$-risky features at $\bar{x}$, and the corresponding sum with the random variables $\tilde{X}_{i\ell}$ is thus an upper bound on the number of risky features at $\bar{x}$. Notice that $X_{i\ell}$ and $\tilde{X}_{i\ell}$ are $\mathcal{F}^\ell$-measurable, and additionally notice that on $\mathcal{G}$, we have $X_{i\ell}/2 \leq \tilde{X}_{i\ell} \leq 2X_{i\ell}$. For any $K \in \{0, 1, \ldots, nL-1, nL\}$, we have by disjointness of the events in the union and a partition

$$\mathbb{P}\left[\sum_{i,\ell} \mathbb{1}_{X_{i\ell} \leq \Delta} > K\right] \leq L^2 e^{-cn/L} + \sum_{k=K+1}^{nL} \mathbb{P}\left[\mathcal{G} \cap \left\{\sum_{i,\ell} \mathbb{1}_{X_{i\ell} \leq \Delta} = k\right\}\right]$$

$$\leq L^2 e^{-cn/L} + \sum_{k=K+1}^{nL} \mathbb{P}\left[\mathcal{G} \cap \left\{\sum_{i,\ell} \mathbb{1}_{\tilde{X}_{i\ell} \leq 2\Delta} = k\right\}\right],$$

so it is essentially equivalent to consider the $\tilde{X}_{i\ell}$. By another partitioning we can write

$$\mathbb{P}\left[\mathcal{G} \cap \left\{\sum_{i,\ell} \mathbb{1}_{\tilde{X}_{i\ell} \leq 2\Delta} = k\right\}\right] = \sum_{S \in \{0,1\}^{n \times L} : \|S\|_F^2 = k} \mathbb{E}\left[\prod_{\ell=1}^{L}\left(\mathbb{1}_{\mathcal{G}_{\ell-1}}\prod_{i=1}^{n}\mathbb{1}_{\tilde{X}_{i\ell} \leq 2\Delta = S_{i\ell}}\right)\right]$$

where $\{0,1\}^{n \times L}$ is the set of $n \times L$ matrices with entries in $\{0,1\}$. Using the tower rule and $\mathcal{F}^{L-1}$-measurability of all factors with $\ell < L$, we can then write

$$\mathbb{E}\left[\prod_{\ell=1}^{L}\mathbb{1}_{\mathcal{G}_{\ell-1}}\prod_{i=1}^{n}\mathbb{1}_{\tilde{X}_{i\ell} \leq 2\Delta = S_{i\ell}}\right]$$

$$= \mathbb{E}\left[\mathbb{E}\left[\prod_{\ell=1}^{L}\left(\mathbb{1}_{\mathcal{G}_{\ell-1}}\prod_{i=1}^{n}\mathbb{1}_{\tilde{X}_{i\ell} \leq 2\Delta = S_{i\ell}}\right) \,\Big|\, \mathcal{F}^{L-1}\right]\right]$$

$$= \mathbb{E}\left[\left(\prod_{\ell=1}^{L-1}\left(\mathbb{1}_{\mathcal{G}_{\ell-1}}\prod_{i=1}^{n}\mathbb{1}_{\tilde{X}_{i\ell} \leq 2\Delta = S_{i\ell}}\right)\right)\mathbb{1}_{\mathcal{G}_{L-1}}\mathbb{E}\left[\prod_{i=1}^{n}\mathbb{1}_{\tilde{X}_{iL} \leq 2\Delta = S_{iL}} \,\Big|\, \mathcal{F}^{L-1}\right]\right].$$

We study the inner conditional expectation as follows: because $\boldsymbol{\rho}^L(\bar{\boldsymbol{x}}) = \boldsymbol{W}^L \boldsymbol{\alpha}^{L-1}(\bar{\boldsymbol{x}})$, we can apply rotational invariance in the conditional expectation to obtain

$$\mathbb{1}_{\mathcal{G}_{L-1}} \mathbb{E}\left[\prod_{i=1}^n \mathbb{1}_{\mathbb{1}_{\tilde{X}_{iL} \le 2\Delta} = S_{iL}} \,\middle|\, \mathcal{F}^{L-1}\right] = \mathbb{1}_{\mathcal{G}_{L-1}} \mathbb{E}\left[\prod_{i=1}^n \mathbb{1}_{\mathbb{1}_{|(\boldsymbol{w}_L)_i|\mathbb{1}_{\mathcal{G}_{\ell-1}} \le 2\Delta} = S_{iL}} \,\middle|\, \mathcal{F}^{L-1}\right]$$
$$= \mathbb{1}_{\mathcal{G}_{L-1}} \mathbb{E}\left[\prod_{i=1}^n \mathbb{1}_{\mathbb{1}_{|(\boldsymbol{w}_L)_i| \le 2\Delta} = S_{iL}} \,\middle|\, \mathcal{F}^{L-1}\right],$$

where $\boldsymbol{w}_L \sim \mathcal{N}(\boldsymbol{0}, (2/n)\boldsymbol{I})$ is the first column of $\boldsymbol{W}^L$, and the last equality takes advantage of the presence of the indicator for $\mathbb{1}_{\mathcal{G}_{\ell-1}}$ multiplying the conditional expectation. We then write using independence

$$\mathbb{E}\left[\prod_{i=1}^n \mathbb{1}_{\mathbb{1}_{|(\boldsymbol{w}_L)_i| \le 2\Delta} = S_{iL}} \,\middle|\, \mathcal{F}^{L-1}\right] = \mathbb{P}\left[\bigcap_{i=1}^n \{\mathbb{1}_{|(\boldsymbol{w}_L)_i| \le 2\Delta} = S_{iL}\} \,\middle|\, \mathcal{F}^{L-1}\right]$$
$$= \prod_{i=1}^n \mathbb{P}\left[\mathbb{1}_{|(\boldsymbol{w}_L)_i| \le 2\Delta} = S_{iL} \,\middle|\, \mathcal{F}^{L-1}\right],$$

and putting $p_L = \mathbb{P}[|(\boldsymbol{w}_L)_1| \le 2\Delta]$, we have by identically-distributedness

$$\prod_{i=1}^n \mathbb{P}\left[\mathbb{1}_{|(\boldsymbol{w}_L)_i| \le 2\Delta} = S_{iL} \,\middle|\, \mathcal{F}^{L-1}\right] = \prod_{i=1}^n p_L^{S_{iL}} (1 - p_L)^{1 - S_{iL}}.$$

After removing the indicator for $\mathcal{G}_{L-1}$ by nonnegativity of all factors in the expectation, this leaves us with

$$\mathbb{E}\left[\prod_{\ell=1}^L \mathbb{1}_{\mathcal{G}_{\ell-1}} \prod_{i=1}^n \mathbb{1}_{\tilde{X}_{i\ell} \le 2\Delta = S_{i\ell}}\right] \le \left(\prod_{i=1}^n p_L^{S_{iL}} (1 - p_L)^{1 - S_{iL}}\right) \mathbb{E}\left[\left(\prod_{\ell=1}^{L-1} \mathbb{1}_{\mathcal{G}_{\ell-1}} \prod_{i=1}^n \mathbb{1}_{\tilde{X}_{i\ell} \le 2\Delta = S_{i\ell}}\right)\right].$$

This process can evidently be iterated $L-1$ additional times with analogous definitions—we observe that the fact that all weight matrices $\boldsymbol{W}^\ell$ have the same column distribution implies that $p_1 = \cdots = p_L$, so we write $p = p_1$ henceforth—and by this we obtain

$$\mathbb{E}\left[\prod_{\ell=1}^L \mathbb{1}_{\mathcal{G}_{\ell-1}} \prod_{i=1}^n \mathbb{1}_{\tilde{X}_{i\ell} \le 2\Delta = S_{i\ell}}\right] \le \prod_{\ell=1}^L \prod_{i=1}^n p^{S_{iL}} (1 - p)^{1 - S_{iL}},$$

and in particular

$$\mathbb{P}\left[\mathcal{G} \cap \left\{\sum_{i,\ell} \mathbb{1}_{\tilde{X}_{i\ell} \le 2\Delta} = k\right\}\right] \le \sum_{\boldsymbol{S} \in \{0,1\}^{n \times L} : \|\boldsymbol{S}\|_F = k} \prod_{\ell=1}^L \prod_{i=1}^n p^{S_{iL}} (1 - p)^{1 - S_{iL}}.$$

For $i \in [n]$ and $\ell \in [L]$, let $Y_{i\ell}$ denote $nL$ i.i.d. $\mathrm{Bern}(p)$ random variables; we recognize this last sum as the probability that $\sum_{i,\ell} Y_{i\ell} = k$. In particular, using our previous work we can assert for any $t > 0$

$$\mathbb{P}\left[\sum_{i,\ell} \mathbb{1}_{X_{i\ell} \le \Delta} > t\right] \le \mathbb{P}\left[\sum_{i,\ell} Y_{i\ell} > t\right] + L^2 e^{-cn/L},$$

so to conclude it suffices to articulate some binomial tail probabilities and estimate $p$. We have

$$p = \mathbb{P}[|(\boldsymbol{w}_L)_i| \le 2\Delta] = \sqrt{\frac{n}{2\pi}} \int_0^{2\Delta} \exp\left(-\frac{nt^2}{4}\right) \mathrm{d}t$$
$$\le \Delta\sqrt{\frac{2n}{\pi}}, \tag{D.54}$$

and we can write with the triangle inequality and a union bound

$$\mathbb{P}\left[\sum_{i,\ell} Y_{i\ell} > t\right] \le \mathbb{P}\left[\left|\sum_{i,\ell} Y_{i\ell} - \mathbb{E}[Y_{i\ell}]\right| > t\right] + \mathbb{P}\left[\sum_{i,\ell} \mathbb{E}[Y_{i\ell}] > t\right].$$

By (D.54), we have $\sum_{i,\ell} \mathbb{E}[Y_{i\ell}] \leq n^{3/2} L \Delta$. We calculate using independence

$$
\mathbb{E}\left[\left(\sum_{i,\ell} Y_{i\ell} - \mathbb{E}[Y_{i\ell}]\right)^2\right] \leq \sum_{i,\ell} \mathbb{E}[Y_{i\ell}] \leq n^{3/2} L \Delta,
$$

so an application of Lemma G.3 yields

$$
\mathbb{P}\left[\left|\sum_{i,\ell} Y_{i\ell} - \mathbb{E}[Y_{i\ell}]\right| > t\right] \leq 2\exp\left(-\frac{t^2/2}{n^{3/2} L \Delta + t/3}\right).
$$

For any $d > 0$, if we choose $t = d \log n$ and enforce $\Delta \leq d \log n / (6n^{3/2} L)$, we obtain

$$
\mathbb{P}\left[\sum_{i,\ell} Y_{i\ell} > d \log n\right] \leq 2n^{-d},
$$

from which we conclude as sought

$$
\mathbb{P}\left[\sum_{i,\ell} \mathbb{1}_{X_{i\ell} \leq \Delta} > d \log n\right] \leq 2n^{-d} + L^2 e^{-cn/L}.
$$

$\square$

The next task is to study the stable sign condition at a point $\bar{x}$ as a function of $\varepsilon$ and $\Delta$, assuming $\Delta$ at least satisfies the hypotheses of Lemma D.5. In particular, we will be interested in conditions under which we can guarantee that $\mathsf{SSC}(\ell - 1)$ holding implies that $\mathsf{SSC}(\ell)$ holds. Let $S_\ell(\bar{x}, \Delta) = [n] \setminus R_\ell(\bar{x}, \Delta)$ denote the $\Delta$-stable features at level $\ell$ with input $\bar{x}$, and define for $0 \leq \ell' \leq \ell \leq L$

$$
\begin{aligned}
\mathbf{T}_{\boldsymbol{x}}^{\ell,\ell'} &= \boldsymbol{P}_{S_\ell(\bar{\boldsymbol{x}})} \boldsymbol{P}_{I_\ell(\boldsymbol{x})} \boldsymbol{W}^\ell \boldsymbol{P}_{S_{\ell-1}(\bar{\boldsymbol{x}})} \boldsymbol{P}_{I_{\ell-1}(\boldsymbol{x})} \boldsymbol{W}^{\ell-1} \ldots \boldsymbol{P}_{S_{\ell'+1}(\bar{\boldsymbol{x}})} \boldsymbol{P}_{I_{\ell'+1}(\boldsymbol{x})} \boldsymbol{W}^{\ell'+1}; \\
\boldsymbol{\Phi}_{\boldsymbol{x}}^{\ell,\ell'} &= \boldsymbol{W}^\ell \mathbf{T}_{\boldsymbol{x}}^{\ell-1,\ell'},
\end{aligned}
\tag{D.55}
$$

so that $\boldsymbol{\Phi}_{\boldsymbol{x}}^{\ell,\ell'} \boldsymbol{x}$ carries an input $\boldsymbol{x} \in \mathcal{N}_\varepsilon(\bar{\boldsymbol{x}})$ applied at the features at level $\ell'$ (in particular, $\ell' = 0$ corresponds to $\boldsymbol{\alpha}^0(\boldsymbol{x}) = \boldsymbol{x}$, the network input) to the preactivations at level $\ell$ in a network restricted to only the stable features at $\bar{\boldsymbol{x}}$. We can write

$$
\begin{aligned}
\boldsymbol{\rho}^\ell(\boldsymbol{x}) &= \boldsymbol{W}^\ell \boldsymbol{P}_{I_{\ell-1}(\boldsymbol{x})} \boldsymbol{W}^{\ell-1} \ldots \boldsymbol{P}_{I_1(\boldsymbol{x})} \boldsymbol{W}^1 \boldsymbol{x}; \\
\boldsymbol{\alpha}^\ell(\boldsymbol{x}) &= \boldsymbol{P}_{I_\ell(\boldsymbol{x})} \boldsymbol{W}^\ell \boldsymbol{P}_{I_{\ell-1}(\boldsymbol{x})} \boldsymbol{W}^{\ell-1} \ldots \boldsymbol{P}_{I_1(\boldsymbol{x})} \boldsymbol{W}^1 \boldsymbol{x},
\end{aligned}
$$

which gives us a useful representation if we disregard all levels with no risky features: let $r = \sum_{\ell=1}^L \mathbb{1}_{|R_\ell(\bar{\boldsymbol{x}}, \Delta)| > 0}$ be the number of levels in the network with risky features, and let $\ell_1 < \ell_2 < \cdots < \ell_r$ denote the levels at which risky features occur. If no risky features occur at a level $\ell$, we of course have $\boldsymbol{P}_{S_\ell(\bar{\boldsymbol{x}})} = \boldsymbol{I}$. Assume to begin that $\ell > \ell_r$, and start by writing

$$
\boldsymbol{\rho}^\ell(\boldsymbol{x}) = \boldsymbol{\Phi}_{\boldsymbol{x}}^{\ell,\ell_r}\left(\boldsymbol{P}_{S_{\ell_r}(\bar{\boldsymbol{x}})} + \boldsymbol{P}_{R_{\ell_r}(\bar{\boldsymbol{x}})}\right) \boldsymbol{P}_{I_{\ell_r}(\boldsymbol{x})} \boldsymbol{\Phi}_{\boldsymbol{x}}^{\ell_r,\ell_r-1}\left(\boldsymbol{P}_{S_{\ell_r-1}(\bar{\boldsymbol{x}})} + \boldsymbol{P}_{R_{\ell_r-1}(\bar{\boldsymbol{x}})}\right)\boldsymbol{P}_{I_{\ell_r-1}(\boldsymbol{x})} \cdots
$$
$$
\cdots \boldsymbol{\Phi}_{\boldsymbol{x}}^{\ell_2,\ell_1}\left(\boldsymbol{P}_{S_{\ell_1}(\bar{\boldsymbol{x}})} + \boldsymbol{P}_{R_{\ell_1}(\bar{\boldsymbol{x}})}\right)\boldsymbol{P}_{I_{\ell_1}(\boldsymbol{x})} \boldsymbol{\Phi}_{\boldsymbol{x}}^{\ell_1,0}.
$$

Now we distribute from left to right, and recombine everything to right on the term corresponding to the projection onto the risky features at $\ell_r$; this gives

$$
\boldsymbol{\rho}^\ell(\boldsymbol{x}) = \boldsymbol{\Phi}_{\boldsymbol{x}}^{\ell,\ell_r} \boldsymbol{P}_{R_{\ell_r}(\bar{\boldsymbol{x}})} \boldsymbol{\alpha}^{\ell_r}(\boldsymbol{x}) + \boldsymbol{\Phi}_{\boldsymbol{x}}^{\ell,\ell_r-1}\left(\boldsymbol{P}_{S_{\ell_r-1}(\bar{\boldsymbol{x}})} + \boldsymbol{P}_{R_{\ell_r-1}(\bar{\boldsymbol{x}})}\right)\boldsymbol{P}_{I_{\ell_r-1}(\boldsymbol{x})} \cdots
$$
$$
\cdots \boldsymbol{\Phi}_{\boldsymbol{x}}^{\ell_2,\ell_1}\left(\boldsymbol{P}_{S_{\ell_1}(\bar{\boldsymbol{x}})} + \boldsymbol{P}_{R_{\ell_1}(\bar{\boldsymbol{x}})}\right)\boldsymbol{P}_{I_{\ell_1}(\boldsymbol{x})} \boldsymbol{\Phi}_{\boldsymbol{x}}^{\ell_1,0}.
$$

We can write

$$
\boldsymbol{\Phi}_{\boldsymbol{x}}^{\ell,\ell_r} \boldsymbol{P}_{R_{\ell_r}(\bar{\boldsymbol{x}})} \boldsymbol{\alpha}^{\ell_r}(\boldsymbol{x}) = \boldsymbol{\Phi}_{\boldsymbol{x}}^{\ell,\ell_r}\big|_{R_{\ell_r}(\bar{\boldsymbol{x}})} \boldsymbol{\alpha}^{\ell_r}(\boldsymbol{x})\big|_{R_{\ell_r}(\bar{\boldsymbol{x}})},
$$

where the restriction notation emphasizes that we are considering a column submatrix of the transfer operator induced by the risky features. Iterating the previous argument, we obtain

$$\boldsymbol{\rho}^\ell(\boldsymbol{x}) = \boldsymbol{\Phi}_{\boldsymbol{x}}^{\ell,0}\boldsymbol{x} + \sum_{i=1}^{r} \boldsymbol{\Phi}_{\boldsymbol{x}}^{\ell,\ell_i}\big|_{R_{\ell_i}(\bar{\boldsymbol{x}})} \boldsymbol{\alpha}^{\ell_i}(\boldsymbol{x})\big|_{R_{\ell_i}(\bar{\boldsymbol{x}})}.$$

It is clear that an analogous argument can be used in the case where $\ell \leq \ell_r$ by adapting which risky features can be visited: we can thus assert

$$\boldsymbol{\rho}^\ell(\boldsymbol{x}) = \boldsymbol{\Phi}_{\boldsymbol{x}}^{\ell,0}\boldsymbol{x} + \sum_{i\in[r]\,:\,\ell_i<\ell} \boldsymbol{\Phi}_{\boldsymbol{x}}^{\ell,\ell_i}\big|_{R_{\ell_i}(\bar{\boldsymbol{x}})} \boldsymbol{\alpha}^{\ell_i}(\boldsymbol{x})\big|_{R_{\ell_i}(\bar{\boldsymbol{x}})}. \tag{D.56}$$

Furthermore, we note that under $\mathsf{SSC}(\ell-1)$, no stable feature supports change on $\mathcal{N}_\varepsilon(\bar{\boldsymbol{x}})$, and so one has for every $\boldsymbol{x} \in \mathcal{N}_\varepsilon(\bar{\boldsymbol{x}})$

$$\boldsymbol{\Phi}_{\bar{\boldsymbol{x}}}^{\ell,\ell'} = \boldsymbol{\Phi}_{\boldsymbol{x}}^{\ell,\ell'},$$

so under $\mathsf{SSC}(\ell-1)$ we have by (D.56)

$$\boldsymbol{\rho}^\ell(\boldsymbol{x}) - \boldsymbol{\rho}^\ell(\bar{\boldsymbol{x}}) = \boldsymbol{\Phi}_{\bar{\boldsymbol{x}}}^{\ell,0}\,(\boldsymbol{x}-\bar{\boldsymbol{x}}) + \sum_{i\in[r]\,:\,\ell_i<\ell} \boldsymbol{\Phi}_{\bar{\boldsymbol{x}}}^{\ell,\ell_i}\Big|_{R_{\ell_i}(\bar{\boldsymbol{x}})} \left(\boldsymbol{\alpha}^{\ell_i}(\boldsymbol{x})\big|_{R_{\ell_i}(\bar{\boldsymbol{x}})} - \boldsymbol{\alpha}^{\ell_i}(\bar{\boldsymbol{x}})\big|_{R_{\ell_i}(\bar{\boldsymbol{x}})}\right).$$

The ReLU $[\cdot]_+$ is 1-Lipschitz with respect to $\|\cdot\|_\infty$, and by monotonicity of the max under restriction and $\mathsf{SSC}(\ell-1)$ we have

$$\left\|\boldsymbol{\alpha}^{\ell_i}(\boldsymbol{x})\big|_{R_{\ell_i}(\bar{\boldsymbol{x}})} - \boldsymbol{\alpha}^{\ell_i}(\bar{\boldsymbol{x}})\big|_{R_{\ell_i}(\bar{\boldsymbol{x}})}\right\|_\infty \leq \left\|\boldsymbol{\rho}^{\ell_i}(\boldsymbol{x})\big|_{R_{\ell_i}(\bar{\boldsymbol{x}})} - \boldsymbol{\rho}^{\ell_i}(\bar{\boldsymbol{x}})\big|_{R_{\ell_i}(\bar{\boldsymbol{x}})}\right\|_\infty$$
$$\leq \left\|\boldsymbol{\rho}^{\ell_i}(\boldsymbol{x}) - \boldsymbol{\rho}^{\ell_i}(\bar{\boldsymbol{x}})\right\|_\infty \leq \Delta.$$

Thus, by the triangle inequality, we have under $\mathsf{SSC}(\ell-1)$ a bound

$$\left\|\boldsymbol{\rho}^\ell(\boldsymbol{x}) - \boldsymbol{\rho}^\ell(\bar{\boldsymbol{x}})\right\|_\infty \leq \varepsilon\left\|\boldsymbol{\Phi}_{\bar{\boldsymbol{x}}}^{\ell,0}\right\|_{\ell^2\to\ell^\infty} + \Delta \sum_{i\in[r]\,:\,\ell_i<\ell} \left\|\boldsymbol{\Phi}_{\bar{\boldsymbol{x}}}^{\ell,\ell_i}\big|_{R_{\ell_i}(\bar{\boldsymbol{x}})}\right\|_{\ell^\infty\to\ell^\infty}. \tag{D.57}$$

This suggests an inductive approach to establishing $\mathsf{SSC}(\ell)$ provided we have established it at previous layers—we just need to control the transfer coefficients in (D.57).

**Lemma D.6.** *There are absolute constants $c, c', C, C', C'', C''' > 0$ and absolute constants $K, K' > 0$ such that for any $1 \leq \ell' < \ell \leq L$, any $d \geq K\log n$ and any $\bar{\boldsymbol{x}} \in \mathbb{S}^{n_0-1}$, if $\Delta \leq cn^{-5/2}$ and $n \geq K'\max\{d^4L, 1\}$, then one has*

$$\mathbb{P}\left[\left\|\boldsymbol{\Phi}_{\bar{\boldsymbol{x}}}^{\ell,0}\right\|_{\ell^2\to\ell^\infty} \leq C(1 + \sqrt{\tfrac{n_0}{n}})\right] \geq 1 - C''e^{-cd},$$

*and for any fixed $S \subset [n]$, one has*

$$\mathbb{P}\left[\left\|\boldsymbol{\Phi}_{\bar{\boldsymbol{x}}}^{\ell,\ell'}\big|_S\right\|_{\ell^\infty\to\ell^\infty} \leq C'|S|\sqrt{\tfrac{d}{n}}\right] \geq 1 - C'''e^{-c'd}.$$

*Proof.* We will use Lemmas D.14 and D.23 to bound the transfer coefficients, so let us first verify the hypotheses of these lemmas. In our setting, the transfer matrices differ only from the 'nominal' transfer matrices by restriction to the stable features at $\bar{\boldsymbol{x}}$; we have $S_\ell(\bar{\boldsymbol{x}}) \cap I_\ell(\bar{\boldsymbol{x}}) = [n] \setminus R_\ell(\bar{\boldsymbol{x}})$, which is an admissible support random variable for Lemmas D.14 and D.23, and in particular

$$\left\|\left(\boldsymbol{P}_{S_\ell(\bar{\boldsymbol{x}})}\boldsymbol{P}_{I_\ell(\bar{\boldsymbol{x}})} - \boldsymbol{P}_{I_\ell(\bar{\boldsymbol{x}})}\right)\boldsymbol{\rho}^\ell(\bar{\boldsymbol{x}})\right\|_2 = \left\|\boldsymbol{P}_{R_\ell(\bar{\boldsymbol{x}})}\boldsymbol{\alpha}^\ell(\bar{\boldsymbol{x}})\right\|_2 \leq \sqrt{n}\Delta$$

by Lemma G.10 and the definition of $R_\ell(\bar{\boldsymbol{x}})$. Additionally, using Lemma D.5, we have if $d \geq 1$, $n \geq KdL$, and $\Delta \leq cd/n^{5/2}$, there is an event $\mathcal{E}$ with measure at least $1 - 2e^{-d} - L^2e^{-cn/L}$ on which there are no more than $d$ risky features at $\bar{\boldsymbol{x}}$. Worsening constants in the scalings of $n$ if necessary and requiring moreover $d \geq K'\log n$ and $n \geq K''d^4$, it follows that we can invoke Lemmas D.14 and D.23 to bound the probability of events involving transfer coefficients multiplied by $\mathbb{1}_\mathcal{E}$. Let us also check the residuals we will obtain when applying Lemma D.23: in the notation there, the vector $\boldsymbol{d}$ has as its $\ell$-th entry $R_\ell(\bar{\boldsymbol{x}})$, and so we have bounds $\|\boldsymbol{d}\|_{1/2} \leq \|\boldsymbol{d}\|_1^2$ and $\|\boldsymbol{d}\|_1 = \sum_\ell R_\ell(\bar{\boldsymbol{x}})$,

which means on the event $\mathcal{E}$, the residual is dominated by the $C\sqrt{d^4 nL}$ term in the scalings we have assumed.

The $\ell^2 \to \ell^\infty$ operator norm of a matrix is the maximum $\ell^2$ norm of a row of the matrix, and the $\ell^\infty \to \ell^\infty$ operator norm is the maximum $\ell^1$ norm of a row. Thus

$$\left\| \boldsymbol{\Phi}_{\bar{\boldsymbol{x}}}^{\ell,0} \right\|_{\ell^2 \to \ell^\infty} = \max_{i=1,\ldots,n} \left\| \boldsymbol{e}_i^* \boldsymbol{\Phi}_{\bar{\boldsymbol{x}}}^{\ell,0} \right\|_2$$

$$= \max_{i=1,\ldots,n} \left\| (\boldsymbol{W}^\ell)_i^* \boldsymbol{T}_{\bar{\boldsymbol{x}}}^{\ell-1,0} \right\|_2$$

where $(\boldsymbol{W}^\ell)_i^*$ is the $i$-th row of $\boldsymbol{W}^\ell$, which is $n_0$-dimensional when $\ell = 1$ and $n$-dimensional otherwise. In particular, we have

$$\left\| \boldsymbol{\Phi}_{\bar{\boldsymbol{x}}}^{1,0} \right\|_{\ell^2 \to \ell^\infty} = \max_{i=1,\ldots,n} \left\| (\boldsymbol{W}^1)_i \right\|_2,$$

and so taking a square root and applying Lemma G.2 and independence of the rows of $\boldsymbol{W}^1$, we have

$$\mathbb{P}\left[ \left\| \boldsymbol{\Phi}_{\bar{\boldsymbol{x}}}^{1,0} \right\|_{\ell^2 \to \ell^\infty} \le 2(1 + \sqrt{\tfrac{n_0}{n}}) \right] \ge 1 - 2ne^{-cn},$$

for $c > 0$ an absolute constant. When $\ell > 1$, we can write

$$\max_{i=1,\ldots,n} \left\| (\boldsymbol{W}^\ell)_i^* \boldsymbol{T}_{\bar{\boldsymbol{x}}}^{\ell-1,0} \right\|_2 = \max_{i=1,\ldots,n} \left\| (\boldsymbol{W}^\ell)_i^* \boldsymbol{T}_{\bar{\boldsymbol{x}}}^{\ell-1,1} \boldsymbol{P}_{I_1(\bar{\boldsymbol{x}})} \boldsymbol{W}^1 \right\|_2$$

$$\le \left\| \boldsymbol{W}^1 \right\| \max_{i=1,\ldots,n} \left\| (\boldsymbol{W}^\ell)_i^* \boldsymbol{T}_{\bar{\boldsymbol{x}}}^{\ell-1,1} \boldsymbol{P}_{I_1(\bar{\boldsymbol{x}})} \right\|_2,$$

where the second line applies Cauchy-Schwarz. Using, say, rotational invariance, Gauss-Lipschitz concentration, and Lemma E.48 (or (Vershynin, 2018, Theorem 4.4.5)), we have

$$\mathbb{P}\left[ \left\| \boldsymbol{W}^1 \right\| > C(1 + \sqrt{\tfrac{n_0}{n}}) \right] \le 2e^{-cn}$$

for absolute constants $c, C > 0$. On the other hand, note that $\|(\boldsymbol{W}^\ell)_i^* \boldsymbol{T}_{\bar{\boldsymbol{x}}}^{\ell-1,1} \boldsymbol{P}_{I_1(\bar{\boldsymbol{x}})}\|_2$ has the same distribution as the square root of one of the index-0 diagonal terms studied in Lemma D.23 in a network truncated at level $\ell - 1$ instead of $L$ and scaled by $2/n$; and so applying this result together with a union bound and the choice $n \ge \max\{K, K'd^4 L\}$ for absolute constants $K, K' > 0$ gives

$$\mathbb{P}\left[ \mathbb{1}_{\mathcal{E}} \max_{i=1,\ldots,n} \left\| (\boldsymbol{W}^\ell)_i^* \boldsymbol{T}_{\bar{\boldsymbol{x}}}^{\ell-1,1} \boldsymbol{P}_{I_1(\bar{\boldsymbol{x}})} \right\|_2 > C' \right] \le C'' n e^{-c'd}$$

where $C', c', C'' > 0$ are absolute constants. We conclude by another union bound

$$\mathbb{P}\left[ \left\| \boldsymbol{\Phi}_{\bar{\boldsymbol{x}}}^{\ell,0} \right\|_{\ell^2 \to \ell^\infty} \le C(1 + \sqrt{\tfrac{n_0}{n}}) \right] \ge 1 - 2e^{-cn} - C'n e^{-c'd} - C'' L^2 e^{-c''n/L}.$$

We can reduce the study of the partial risky propagation coefficients to a similar calculation. We have

$$\left\| \boldsymbol{\Phi}_{\bar{\boldsymbol{x}}}^{\ell,\ell'} \big|_S \right\|_{\ell^\infty \to \ell^\infty} = \max_{j=1,\ldots,n} \left\| (\boldsymbol{W}^\ell)_j^* \left( \boldsymbol{T}_{\bar{\boldsymbol{x}}}^{\ell-1,\ell'} \big|_S \right) \right\|_1,$$

where by construction we have that $\ell > \ell'$. In the case $\ell = \ell' + 1$, the form is slightly different; we have

$$\left\| \boldsymbol{\Phi}_{\bar{\boldsymbol{x}}}^{\ell'+1,\ell'} \big|_S \right\|_{\ell^\infty \to \ell^\infty} = \max_{j=1,\ldots,n} \left\| \left( \boldsymbol{W}^{\ell'+1} \big|_S \right)_j \right\|_1 \le |S| \max_{j=1,\ldots,n} \left\| \left( \boldsymbol{W}^{\ell'+1} \big|_S \right)_j \right\|_\infty,$$

where the inequality uses Lemma G.10. The classical estimate for the gaussian tail gives

$$\mathbb{P}\left[ \left| (\boldsymbol{W}^{\ell'+1})_{jk} \right| > \sqrt{\tfrac{2d}{n}} \right] \le 2e^{-d}, \tag{D.58}$$

for each $k \in [n]$, so a union bound gives

$$\mathbb{P}\left[ \max_{j=1,\ldots,n} \left\| \left( \boldsymbol{W}^{\ell'+1} \big|_S \right)_j \right\|_\infty > \sqrt{\tfrac{2d}{n}} \right] \le 2ne^{-d},$$

and we conclude

$$\mathbb{P}\left[\left\|\left.\mathbf{\Phi}_{\bar{\boldsymbol{x}}}^{\ell'+1,\ell'}\right|_{S}\right\|_{\ell^{\infty}\to\ell^{\infty}} \le |S|\sqrt{\frac{2d}{n}}\right] \ge 1 - 2e^{-d/2},$$

where the final bound holds if $d \ge 2\log n$. Next, we assume $\ell > \ell' + 1$. In this case, Lemma G.10 gives

$$\left\|\left.\mathbf{\Phi}_{\bar{\boldsymbol{x}}}^{\ell,\ell'}\right|_{S}\right\|_{\ell^{\infty}\to\ell^{\infty}} = \max_{j=1,\dots,n}\left\|(\boldsymbol{W}^{\ell})_{j}^{*}\left(\left.\mathbf{T}_{\bar{\boldsymbol{x}}}^{\ell-1,\ell'}\right|_{S}\right)\right\|_{1}$$

$$\le |S|\max_{j=1,\dots,n}\left\|(\boldsymbol{W}^{\ell})_{j}^{*}\left(\left(\left.\mathbf{T}_{\bar{\boldsymbol{x}}}^{\ell-1,\ell'}\right|_{S}\right)\right)\right\|_{\infty}.$$

For the second term on the RHS of the inequality, we write

$$\max_{j=1,\dots,n}\left\|(\boldsymbol{W}^{\ell})_{j}^{*}\left.\mathbf{T}_{\bar{\boldsymbol{x}}}^{\ell-1,\ell'}\right|_{S}\right\|_{\infty} = \max_{j=1,\dots,n}\max_{k\in S}\left|(\boldsymbol{W}^{\ell})_{j}^{*}\mathbf{T}_{\bar{\boldsymbol{x}}}^{\ell-1,\ell'}\boldsymbol{e}_{k}\right|$$

then apply rotational invariance of the distribution of $(\boldsymbol{W}^{\ell})_{j}$ and $\mathcal{F}^{\ell-1}$-measurability of $\left.\mathbf{T}_{\bar{\boldsymbol{x}}}^{\ell-1,\ell'}\right|_{S}$ to obtain

$$\max_{j=1,\dots,n}\max_{k\in S}\left|(\boldsymbol{W}^{\ell})_{j}^{*}\mathbf{T}_{\bar{\boldsymbol{x}}}^{\ell-1,\ell'}\boldsymbol{e}_{k}\right| = \max_{j=1,\dots,n}\max_{k\in S\,:\,\left\|\mathbf{T}_{\bar{\boldsymbol{x}}}^{\ell-1,\ell'}\boldsymbol{e}_{k}\right\|_{2}>0}\left|(\boldsymbol{W}^{\ell})_{j}^{*}\mathbf{T}_{\bar{\boldsymbol{x}}}^{\ell-1,\ell'}\boldsymbol{e}_{k}\right|$$

$$\overset{d}{=} \max_{j=1,\dots,n}\max_{k\in S}|g_{j}|\left\|\mathbf{T}_{\bar{\boldsymbol{x}}}^{\ell-1,\ell'}\boldsymbol{e}_{k}\right\|_{2},$$

where $\boldsymbol{g} \sim \mathcal{N}(\mathbf{0},(2/n)\boldsymbol{I})$ is independent of everything else in the problem. We have by Lemma D.14 based on our previous choices of $n$ and $d$

$$\mathbb{P}\left[\mathbb{1}_{\mathcal{E}}\left\|\mathbf{T}_{\bar{\boldsymbol{x}}}^{\ell-1,\ell'}\boldsymbol{e}_{k}\right\|_{2} \le C\right] \ge 1 - C'e^{-cn/L},$$

and (D.58) applied to $\boldsymbol{g}$ controls the remaining term. Taking a union bound over $[n]$ in these two estimates and partitioning with $\mathcal{E}$, we conclude

$$\mathbb{P}\left[\max_{j=1,\dots,n}\left\|(\boldsymbol{W}^{\ell})_{j}^{*}\left(\left.\mathbf{T}_{\bar{\boldsymbol{x}}}^{\ell-1,\ell'}\right|_{S}\right)\right\|_{\infty} > C\sqrt{\frac{d}{n}}\right] \le 2ne^{-d} + C'ne^{-cn/L} + 2e^{-d} + L^{2}e^{-c'n/L}$$

and thus

$$\mathbb{P}\left[\left\|\left.\mathbf{\Phi}_{\bar{\boldsymbol{x}}}^{\ell,\ell'}\right|_{S}\right\|_{\ell^{\infty}\to\ell^{\infty}} > C|S|\sqrt{\frac{d}{n}}\right] \le C'e^{-d/2} + C''n^{2}e^{-cn/L},$$

where the last bound holds if $d \ge 2\log n$. Choosing $n \ge KL\log n$ for a suitable absolute constant $K > 0$ allows us to simplify the residual terms in the probability bounds to the forms we have claimed. $\qquad\square$

**Lemma D.7.** *There is an absolute constant $c > 0$ and absolute constants $k, k', K, K' > 0$ such that for any $d \ge K\log n$, if $n \ge K'\max\{d^{4}L, 1\}$, $\Delta \le kn^{-5/2}$, and $\varepsilon \le k'\Delta\left(1 + \sqrt{\frac{n_{0}}{n}}\right)^{-1}$, then one has for any $\bar{\boldsymbol{x}} \in N_{\varepsilon}$*

$$\mathbb{P}\left[\{\mathsf{SSC}(L) \text{ holds at } \bar{\boldsymbol{x}}\} \cap \left\{\sum_{\ell=1}^{L}|R_{\ell}(\bar{\boldsymbol{x}},\Delta)| \le d\right\}\right] \ge 1 - e^{-cd}.$$

*Proof.* We start by constructing a high-probability event on which we have control of every possible propagation coefficient. For any $d \ge K\log n$, choosing $\Delta \le cn^{-5/2}$ and $n \ge K'd^{4}L$ and applying the first conclusion in Lemma D.6 and a union bound, we have

$$\mathbb{P}\left[\exists \ell \in [L], \ \left\|\mathbf{\Phi}_{\bar{\boldsymbol{x}}}^{\ell,0}\right\|_{\ell^{2}\to\ell^{\infty}} > C_{1}(1 + \sqrt{\frac{n_{0}}{n}})\right] \le Ce^{-cd} \tag{D.59}$$

and under the same hypotheses, for any $1 \le \ell' < \ell \le L$ and any $S \subset [n]$, the second conclusion in Lemma D.6 gives

$$\mathbb{P}\left[\left\|\left.\mathbf{\Phi}_{\bar{\boldsymbol{x}}}^{\ell,\ell'}\right|_{S}\right\|_{\ell^{\infty}\to\ell^{\infty}} > C_{2}|S|\sqrt{\frac{d}{n}}\right] \le C'e^{-16d}.$$

Using Lemma D.5, we have if $n \geq \max\{KdL, 4\}$ and $\Delta \leq K'/n^{5/2}$

$$\mathbb{P}\left[\sum_{\ell=1}^{L} |R_\ell(\bar{\boldsymbol{x}}, \Delta)| > d\right] \leq 2e^{-d} + L^2 e^{-cn/L}. \tag{D.60}$$

Denote the complement of the event in the previous bound as $\mathcal{E}$. On $\mathcal{E}$, there are no more than $d$ levels in the network with risky features. There are at most $\sum_{k=0}^{\lceil d \rceil} \binom{n}{k}$ ways to choose a subset $S \subset [n]$ with cardinality at most $\lceil d \rceil$; using $n \geq e$ and $d \geq 1$, we have

$$\sum_{k=0}^{\lceil d \rceil} \binom{n}{k} \leq 1 + \lceil d \rceil n^{\lceil d \rceil} \leq 4dn^{2d}.$$

In addition, there are at most $L^2$ ways to pick two indices $1 \leq \ell' < \ell \leq L$. Using $n \geq L$, this yields at most $4dn^{2+2d} \leq n^{8d}$ items to union bound over, i.e.,

$$\mathbb{P}\left[\bigcup_{\substack{S \subset [n] \\ |S| \leq \lceil d \rceil}} \bigcup_{1 \leq \ell < \ell' \leq L} \left\{\left\|\boldsymbol{\Phi}_{\bar{\boldsymbol{x}}}^{\ell,\ell'}\big|_S\right\|_{\ell^\infty \to \ell^\infty} > C_2 |S| \sqrt{\frac{d}{n}}\right\}\right] \leq Ce^{-8d} \tag{D.61}$$

if $d \geq K \log n$ and $n \geq \max\{K'd^4 L, n_0\}$. Denote the complement of the union of the events in the bounds (D.59) to (D.61) and $\mathcal{E}$ as $\mathcal{G}$; taking additional union bounds and worst-casing absolute constants, we have shown

$$\mathbb{P}[\mathcal{G}] \geq 1 - Ce^{-cd}.$$

If we enumerate the levels of the network that have risky features as $1 \leq \ell_1 < \cdots < \ell_r \leq L$, it follows from our previous counting argument that on $\mathcal{G}$, we have transfer coefficient bounds (for any $\ell \in [L]$ and any $\ell_i < \ell$)

$$\left\|\boldsymbol{\Phi}_{\bar{\boldsymbol{x}}}^{\ell,0}\right\|_{\ell^2 \to \ell^\infty} \leq C_1\left(1 + \sqrt{\frac{n_0}{n}}\right), \qquad \left\|\boldsymbol{\Phi}_{\bar{\boldsymbol{x}}}^{\ell,\ell_i}\big|_{R_{\ell_i}(\bar{\boldsymbol{x}})}\right\|_{\ell^\infty \to \ell^\infty} \leq C_2 |R_{\ell_i}(\bar{\boldsymbol{x}})| \sqrt{\frac{d}{n}}.$$

Now we begin the induction. Let $\boldsymbol{x} \in \mathcal{N}_\varepsilon(\bar{\boldsymbol{x}})$. For $\mathsf{SSC}(1)$, we have from the definitions

$$\left\|\boldsymbol{\rho}^1(\boldsymbol{x}) - \boldsymbol{\rho}^1(\bar{\boldsymbol{x}})\right\|_\infty \leq \varepsilon \left\|\boldsymbol{\Phi}_{\bar{\boldsymbol{x}}}^{1,0}\right\|_{\ell^2 \to \ell^\infty} \leq C_1\left(1 + \sqrt{\frac{n_0}{n}}\right)\varepsilon,$$

where the last inequality holds on $\mathcal{G}$. So we have $\mathsf{SSC}(1)$ on $\mathcal{G}$ if $\varepsilon \leq \Delta(C_1(1 + \sqrt{\frac{n_0}{n}}))^{-1}$. Continuing, we suppose that we have established $\mathsf{SSC}(\ell - 1)$ on $\mathcal{G}$. We can therefore apply (D.57) together with our transfer coefficient bounds to get

$$\left\|\boldsymbol{\rho}^\ell(\boldsymbol{x}) - \boldsymbol{\rho}^\ell(\bar{\boldsymbol{x}})\right\|_\infty \leq C_1\left(1 + \sqrt{\frac{n_0}{n}}\right)\varepsilon + C_2 \Delta \sqrt{\frac{d}{n}} \sum_{i \in [r]\,:\,\ell_i < \ell} |R_{\ell_i}(\bar{\boldsymbol{x}})|$$

$$\leq C_1\left(1 + \sqrt{\frac{n_0}{n}}\right)\varepsilon + C_2 \Delta \sqrt{\frac{d^3}{n}}.$$

Notice that the last bound does not depend on $\ell$. Thus, if we choose $\varepsilon \leq \Delta(2C_1(1 + \sqrt{\frac{n_0}{n}}))^{-1}$ and $n \geq 4C_2^2 d^3$, we obtain $\left\|\boldsymbol{\rho}^\ell(\boldsymbol{x}) - \boldsymbol{\rho}^\ell(\bar{\boldsymbol{x}})\right\|_\infty \leq \Delta$; by induction, we can conclude that $\mathsf{SSC}(L)$ holds on $\mathcal{G}$, which implies the claim; we obtain the final simplified probability bound by worsening the constant in the exponent. $\qquad\square$

### D.3.3 Uniformizing Forward Features Under SSC

Under the $\mathsf{SSC}(L)$ condition, we can uniformize forward and backward features. A prerequisite of our approach, which we also used to establish $\mathsf{SSC}(L)$ in the previous section, is control of certain residuals that appear when a small number of supports can change off the nominal forward and backward correlations. These estimates are studied in the next section, Appendix D.3.4.

In previous sections, most of our results (e.g. Lemma D.1) feature a lower bound of the type $n \geq K$ in their hypotheses. After uniformizing, we will discard this hypothesis using our extra assumption

that $n_0 \geq 3$, which gives us a lower bound on the logarithmic terms of the form $\log(Cnn_0)$ that appear as lower bounds on $d$ after uniformizing, and the fact that our lower bounds on $n$ always involve a polynomial in $d$. Thus, by adjusting absolute constants, we can achieve the same effect as previously.

**Lemma D.8.** *There are absolute constants $c, C > 0$ and an absolute constant $K, K' > 0$ such that for any $d \geq K d_0 \log(nn_0 C_{\mathcal{M}})$, if $n \geq K' d^4 L$ then one has*

$$\mathbb{P}\left[\bigcap_{\bar{\boldsymbol{x}} \in N_{n^{-3}n_0^{-1/2}}} \left\{\mathsf{SSC}(L, n^{-3}n_0^{-1/2}, Cn^{-3}) \text{ holds at } \bar{\boldsymbol{x}}\right\} \cap \left\{\sum_{\ell=1}^{L} |R_\ell(\bar{\boldsymbol{x}}, Cn^{-3})| \leq d\right\}\right]$$
$$\geq 1 - e^{-cd}.$$

*Proof.* Following the discussion in Appendix D.3.1, if $0 < \varepsilon \leq 1$ then $|N_\varepsilon| \leq e^{d_0 \log(C_{\mathcal{M}}/\varepsilon)}$; to apply Lemma D.7 we at least need $\Delta \leq kn^{-5/2}$ and $\varepsilon \leq k'\Delta\left(1 + \sqrt{\frac{n_0}{n}}\right)^{-1}$, so it suffices to put $\Delta = Cn^{-3}$ when $n$ is chosen larger than an absolute constant, and require $\varepsilon \leq \min\left\{1, k'Cn^{-5/2}\left(1 + \sqrt{\frac{n_0}{n}}\right)^{-1}\right\}$. Fixing $\varepsilon = n^{-3}n_0^{-1/2}$, which again is admissible when $n$ is sufficiently large compared to an absolute constant, for any $d \geq K d_0 \log(nn_0 C_{\mathcal{M}})$, we choose $n \geq K \max\{1, d^4 L\}$ and take a union bound to obtain the claim (using here that $C_{\mathcal{M}} \geq 1$). $\qquad\square$

**Lemma D.9.** *There is an absolute constant $c > 0$ and absolute constants $K, K', K'' > 0$ such that for any $d \geq K d_0 \log(nn_0 C_{\mathcal{M}})$, if $n \geq K d^4 L$, then one has*

$$\mathbb{P}\left[\bigcap_{\boldsymbol{x} \in \mathcal{M}} \left\{\forall \ell \in [L], \big|\|\boldsymbol{\alpha}^\ell(\boldsymbol{x})\|_2 - 1\big| \leq \frac{1}{2}\right\}\right] \geq 1 - e^{-cd}.$$

*Proof.* Let $\bar{\boldsymbol{x}} \in N_{n^{-3}n_0^{-1/2}}$. Lemma D.2 and a union bound give

$$\mathbb{P}\left[\exists \ell \in [L] : \big|\|\boldsymbol{\alpha}^\ell(\bar{\boldsymbol{x}})\|_2 - 1\big| > \frac{1}{4}\right] \leq C'L^2 e^{-cn/L} \leq e^{-c'd}$$

if $n \geq KdL$ and $d \geq K' \log n$. If additionally $d \geq K'd_0 \log(nn_0 C_{\mathcal{M}})$, we obtain by the discussion in Appendix D.3.1 and another union bound

$$\mathbb{P}\left[\bigcup_{\bar{\boldsymbol{x}} \in N_{n^{-3}n_0^{-1/2}}} \left\{\exists \ell \in [L] : \big|\|\boldsymbol{\alpha}^\ell(\bar{\boldsymbol{x}})\|_2 - 1\big| > \frac{1}{4}\right\}\right] \leq e^{-cd}.$$

Let $\mathcal{E}$ denote the event studied in Lemma D.8; choose $d \geq K d_0 \log(nn_0 C_{\mathcal{M}})$ and $n$ sufficiently large to make the measure bound applicable. A union bound gives

$$\mathbb{P}\left[\mathcal{E}^{\mathsf{c}} \cup \bigcup_{\bar{\boldsymbol{x}} \in N_{n^{-3}n_0^{-1/2}}} \left\{\exists \ell \in [L] : \big|\|\boldsymbol{\alpha}^\ell(\bar{\boldsymbol{x}})\|_2 - 1\big| > \frac{1}{4}\right\}\right] \leq e^{-cd}.$$

Let $\mathcal{G}$ denote the complement of the event in the previous bound. For any $\boldsymbol{x} \in \mathcal{M}$, we can find a point $\bar{\boldsymbol{x}} \in N_{n^{-3}n_0^{-1/2}} \cap \mathcal{N}_{n^{-3}n_0^{-1/2}}(\boldsymbol{x})$. On $\mathcal{G}$, $\mathsf{SSC}(L, n^{-3}n_0^{-1/2}, Cn^{-3})$ holds at every point in the net $N_{n^{-3}n_0^{-1/2}}$, which implies that on $\mathcal{G}$

$$\forall \ell \in [L], \big\|\boldsymbol{\rho}^\ell(\boldsymbol{x}) - \boldsymbol{\rho}^\ell(\bar{\boldsymbol{x}})\big\|_\infty \leq \frac{C}{n^3}, \tag{D.62}$$

and by the 1-Lipschitz property of $[\,\cdot\,]_+$ and Lemma G.10, this also implies that on $\mathcal{G}$

$$\forall \ell \in [L], \big\|\boldsymbol{\alpha}^\ell(\boldsymbol{x}) - \boldsymbol{\alpha}^\ell(\bar{\boldsymbol{x}})\big\|_2 \leq \frac{C}{n^{5/2}}.$$

Choosing $n \geq (4C)^{2/5}$, the RHS of this bound is no larger than $1/4$. We write using the triangle inequality

$$\left| \|\boldsymbol{\alpha}^\ell(\boldsymbol{x})\|_2 - 1 \right| \leq \left| \|\boldsymbol{\alpha}^\ell(\boldsymbol{x})\|_2 - \|\boldsymbol{\alpha}^\ell(\bar{\boldsymbol{x}})\|_2 \right| + \left| \|\boldsymbol{\alpha}^\ell(\bar{\boldsymbol{x}})\|_2 - 1 \right|.$$

Using the triangle inequality again, the first term on the RHS is no larger than $1/4$ for any $\ell$ on $\mathcal{G}$. The second term is also no larger than $1/4$ on $\mathcal{G}$ by control over the net. We conclude that on $\mathcal{G}$

$$\forall \ell \in [L], \ \left| \|\boldsymbol{\alpha}^\ell(\boldsymbol{x})\|_2 - 1 \right| \leq \frac{1}{2}.$$

This implies that the event $\mathcal{G}$ is contained in the set

$$\bigcap_{\boldsymbol{x} \in \mathcal{M}} \left\{ \forall \ell \in [L], \ \left| \|\boldsymbol{\alpha}^\ell(\boldsymbol{x})\|_2 - 1 \right| \leq \frac{1}{2} \right\},$$

which is closed, by continuity of $\|\cdot\|_2$ and of the features as a function of the parameters, and is therefore also an event. The claim follows. $\qquad \square$

**Lemma D.10.** *There are absolute constants $c, C > 0$ and absolute constants $K, K' > 0$ such that for any $d \geq K d_0 \log(n n_0 C_\mathcal{M})$, if $n \geq K' d^4 L$, then one has*

$$\mathbb{P}\left[ \bigcap_{(\boldsymbol{x}, \boldsymbol{x}') \in \mathcal{M} \times \mathcal{M}} \left\{ \forall \ell \in [L], \ \left| \langle \boldsymbol{\alpha}^\ell(\boldsymbol{x}), \boldsymbol{\alpha}^\ell(\boldsymbol{x}') \rangle - \cos \varphi^{(\ell)}(\angle(\boldsymbol{x}, \boldsymbol{x}')) \right| \leq C\sqrt{\frac{d^3 \ell}{n}} \right\} \right] \geq 1 - e^{-cd}.$$

*Proof.* Let $\bar{\boldsymbol{x}}, \bar{\boldsymbol{x}}' \in N_{n^{-3} n_0^{-1/2}}$. Lemma D.1 and a union bound give

$$\mathbb{P}\left[ \exists \ell \in [L] : \ \left| \langle \boldsymbol{\alpha}^\ell(\bar{\boldsymbol{x}}), \boldsymbol{\alpha}^\ell(\bar{\boldsymbol{x}}') \rangle - \cos \varphi^{(\ell)}(\angle(\bar{\boldsymbol{x}}, \bar{\boldsymbol{x}}')) \right| > C\sqrt{\frac{d^3 \ell}{n}} \right] \leq C' L e^{-cd} \leq e^{-c'd}$$

if $d \geq K \log n$ and $n \geq \max\{K' d^3 L, K'' d^4, K'''\}$. If additionally $d \geq K d_0 \log(n n_0 C_\mathcal{M})$, we obtain by the discussion in Appendix D.3.1 and another union bound

$$\mathbb{P}\left[ \bigcup_{(\bar{\boldsymbol{x}}, \bar{\boldsymbol{x}}') \in N^{\times 2}_{\frac{1}{n^3 \sqrt{n_0}}}} \left\{ \exists \ell \in [L] : \ \left| \langle \boldsymbol{\alpha}^\ell(\bar{\boldsymbol{x}}), \boldsymbol{\alpha}^\ell(\bar{\boldsymbol{x}}') \rangle - \cos \varphi^{(\ell)}(\angle(\bar{\boldsymbol{x}}, \bar{\boldsymbol{x}}')) \right| > C\sqrt{\frac{d^3 \ell}{n}} \right\} \right] \leq e^{-cd},$$

where with an abuse of notation we write $S^{\times 2}$ to denote $S \times S$ for a set $S$. Let $\mathcal{E}_1$ denote the event studied in Lemma D.8, and let $\mathcal{E}_2$ denote the event studied in Lemma D.9; choose $n$ sufficiently large to make the measure bounds applicable. A union bound gives

$$\mathbb{P}\left[ \mathcal{E}_1^\mathsf{c} \cup \mathcal{E}_2^\mathsf{c} \cup \bigcup_{(\bar{\boldsymbol{x}}, \bar{\boldsymbol{x}}') \in N^{\times 2}_{\frac{1}{n^3 \sqrt{n_0}}}} \left\{ \exists \ell \in [L] : \ \left| \langle \boldsymbol{\alpha}^\ell(\bar{\boldsymbol{x}}), \boldsymbol{\alpha}^\ell(\bar{\boldsymbol{x}}') \rangle - \cos \varphi^{(\ell)}(\angle(\bar{\boldsymbol{x}}, \bar{\boldsymbol{x}}')) \right| > C\sqrt{\frac{d^3 \ell}{n}} \right\} \right]$$
$$\leq e^{-cd}$$

after adjusting constants. Let $\mathcal{G}$ denote the complement of the event in the previous bound. For any $(\boldsymbol{x}, \boldsymbol{x}') \in \mathcal{M} \times \mathcal{M}$, we can find a point $\bar{\boldsymbol{x}} \in N_{n^{-3} n_0^{-1/2}} \cap \mathcal{N}_{n^{-3} n_0^{-1/2}}(\boldsymbol{x})$ and a point $\bar{\boldsymbol{x}}' \in N_{n^{-3} n_0^{-1/2}} \cap \mathcal{N}_{n^{-3} n_0^{-1/2}}(\boldsymbol{x}')$. On $\mathcal{G}$, $\mathsf{SSC}(L, n^{-3} n_0^{-1/2}, Cn^{-3})$ holds at every point in the net $N_{n^{-3} n_0^{-1/2}}$, which implies that on $\mathcal{G}$

$$\forall \ell \in [L], \ \left\| \boldsymbol{\rho}^\ell(\boldsymbol{x}) - \boldsymbol{\rho}^\ell(\bar{\boldsymbol{x}}) \right\|_\infty \leq \frac{C}{n^3}, \quad \text{and} \quad \forall \ell \in [L], \ \left\| \boldsymbol{\rho}^\ell(\boldsymbol{x}') - \boldsymbol{\rho}^\ell(\bar{\boldsymbol{x}}') \right\|_\infty \leq \frac{C}{n^3},$$

and by the 1-Lipschitz property of $[\,\cdot\,]_+$ and Lemma G.10, this also implies that on $\mathcal{G}$

$$\forall \ell \in [L], \ \left\| \boldsymbol{\alpha}^\ell(\boldsymbol{x}) - \boldsymbol{\alpha}^\ell(\bar{\boldsymbol{x}}) \right\|_2 \leq \frac{C}{n^{5/2}}, \quad \text{and} \quad \forall \ell \in [L], \ \left\| \boldsymbol{\alpha}^\ell(\boldsymbol{x}') - \boldsymbol{\alpha}^\ell(\bar{\boldsymbol{x}}') \right\|_2 \leq \frac{C}{n^{5/2}}. \quad \text{(D.63)}$$

For any $\ell \in [L]$, we write using the triangle inequality

$$
\begin{aligned}
\left| \langle \boldsymbol{\alpha}^\ell(\boldsymbol{x}), \boldsymbol{\alpha}^\ell(\boldsymbol{x}') \rangle - \cos \varphi^{(\ell)}(\angle(\boldsymbol{x}, \boldsymbol{x}')) \right| \leq & \left| \langle \boldsymbol{\alpha}^\ell(\boldsymbol{x}), \boldsymbol{\alpha}^\ell(\boldsymbol{x}') \rangle - \langle \boldsymbol{\alpha}^\ell(\bar{\boldsymbol{x}}), \boldsymbol{\alpha}^\ell(\boldsymbol{x}') \rangle \right| \\
& + \left| \langle \boldsymbol{\alpha}^\ell(\bar{\boldsymbol{x}}), \boldsymbol{\alpha}^\ell(\boldsymbol{x}') \rangle - \langle \boldsymbol{\alpha}^\ell(\bar{\boldsymbol{x}}), \boldsymbol{\alpha}^\ell(\bar{\boldsymbol{x}}') \rangle \right| \\
& + \left| \langle \boldsymbol{\alpha}^\ell(\bar{\boldsymbol{x}}), \boldsymbol{\alpha}^\ell(\bar{\boldsymbol{x}}') \rangle - \cos \varphi^{(\ell)}(\angle(\bar{\boldsymbol{x}}, \bar{\boldsymbol{x}}')) \right| \\
& + \left| \cos \varphi^{(\ell)}(\angle(\bar{\boldsymbol{x}}, \bar{\boldsymbol{x}}')) - \cos \varphi^{(\ell)}(\angle(\boldsymbol{x}, \boldsymbol{x}')) \right|.
\end{aligned}
\tag{D.64}
$$

Using Cauchy-Schwarz, we have on $\mathcal{G}$

$$
\left| \langle \boldsymbol{\alpha}^\ell(\boldsymbol{x}), \boldsymbol{\alpha}^\ell(\boldsymbol{x}') \rangle - \langle \boldsymbol{\alpha}^\ell(\bar{\boldsymbol{x}}), \boldsymbol{\alpha}^\ell(\boldsymbol{x}') \rangle \right| \leq \left\| \boldsymbol{\alpha}^\ell(\boldsymbol{x}) - \boldsymbol{\alpha}^\ell(\bar{\boldsymbol{x}}) \right\|_2 \left\| \boldsymbol{\alpha}^\ell(\boldsymbol{x}') \right\|_2 \leq \frac{2C}{n^{5/2}},
$$

with the same bound holding for the second term in (D.64) by an analogous argument. For the third term, we have on $\mathcal{G}$

$$
\left| \langle \boldsymbol{\alpha}^\ell(\bar{\boldsymbol{x}}), \boldsymbol{\alpha}^\ell(\bar{\boldsymbol{x}}') \rangle - \cos \varphi^{(\ell)}(\angle(\bar{\boldsymbol{x}}, \bar{\boldsymbol{x}}')) \right| \leq C \sqrt{\frac{d^3 \ell}{n}}.
$$

For the last term, we use 1-Lipschitzness of $\cos$ and 1-Lipschitzness of the $\varphi^{(\ell)}$, which follows from Lemma E.5 and the chain rule, to obtain

$$
\left| \cos \varphi^{(\ell)}(\angle(\bar{\boldsymbol{x}}, \bar{\boldsymbol{x}}')) - \cos \varphi^{(\ell)}(\angle(\boldsymbol{x}, \boldsymbol{x}')) \right| \leq \left| \angle(\bar{\boldsymbol{x}}, \bar{\boldsymbol{x}}') - \angle(\boldsymbol{x}, \boldsymbol{x}') \right|.
$$

Using Lemma C.7 and several applications of the triangle inequality, we get

$$
\begin{aligned}
\left| \angle(\bar{\boldsymbol{x}}, \bar{\boldsymbol{x}}') - \angle(\boldsymbol{x}, \boldsymbol{x}') \right| & \leq \sqrt{2} |\|\boldsymbol{x} - \boldsymbol{x}'\|_2 - \|\bar{\boldsymbol{x}} - \bar{\boldsymbol{x}}'\|_2| \\
& \leq \sqrt{2} \|(\boldsymbol{x} - \boldsymbol{x}') - (\bar{\boldsymbol{x}} - \bar{\boldsymbol{x}}')\|_2 \\
& \leq \sqrt{2} \|\boldsymbol{x} - \bar{\boldsymbol{x}}\|_2 + \sqrt{2} \|\boldsymbol{x}' - \bar{\boldsymbol{x}}'\|_2 \leq \frac{2\sqrt{2}}{n^3},
\end{aligned}
$$

and so returning to (D.64), we have shown

$$
\left| \langle \boldsymbol{\alpha}^\ell(\boldsymbol{x}), \boldsymbol{\alpha}^\ell(\boldsymbol{x}') \rangle - \cos \varphi^{(\ell)}(\angle(\boldsymbol{x}, \boldsymbol{x}')) \right| \leq C \sqrt{\frac{d^3 \ell}{n}} + \frac{C'}{n^{5/2}} \leq (C + C') \sqrt{\frac{d^3 \ell}{n}}
$$

for every $\ell \in [L]$. This implies that the event $\mathcal{G}$ is contained in the set

$$
\bigcap_{(\boldsymbol{x}, \boldsymbol{x}') \in \mathcal{M} \times \mathcal{M}} \left\{ \forall \ell \in [L], \left| \langle \boldsymbol{\alpha}^\ell(\boldsymbol{x}), \boldsymbol{\alpha}^\ell(\boldsymbol{x}') \rangle - \cos \varphi^{(\ell)}(\angle(\boldsymbol{x}, \boldsymbol{x}')) \right| \leq C \sqrt{\frac{d^3 \ell}{n}} \right\},
$$

which is closed, by continuity of the inner product and of the features as a function of the parameters, and is therefore also an event. The claim follows. $\square$

**Lemma D.11.** *Assume $n, L, d$ satisfy the requirements of lemma D.10 and additionally $d \geq 1, n \geq K\sqrt{L}$ for some $K$. Then*

$$
\mathbb{P}\left[ \|f_{\boldsymbol{\theta}_0}\|_{L^\infty} \leq \sqrt{d} \right] \geq 1 - e^{-cd},
$$

$$
\mathbb{P}\left[ \|\zeta\|_{L^\infty} \leq \sqrt{d} \right] \geq 1 - e^{-cd}.
$$

*Define*

$$
\hat{\zeta}(\boldsymbol{x}) = -f_\star(\boldsymbol{x}) + \int_{\mathcal{M}} f_{\boldsymbol{\theta}_0}(\boldsymbol{x}') \mathrm{d}\mu^\infty(\boldsymbol{x}').
$$

*Then under the same assumptions*

$$
\mathbb{P}\left[ \left\| \hat{\zeta} - \zeta \right\|_{L^\infty} \leq \sqrt{\frac{d}{L^2} + d^{5/2} \sqrt{\frac{L}{n}}} \right] \geq 1 - e^{-cd}
$$

*for some numerical constant $c$.*

*Proof.* At some $\boldsymbol{x} \in \mathcal{M}$, we note that

$$f_{\boldsymbol{\theta}_0}(\boldsymbol{x}) = \boldsymbol{W}^{L+1}\boldsymbol{\alpha}^L(\boldsymbol{x}) \overset{d}{=} g \left\|\boldsymbol{\alpha}^L(\boldsymbol{x})\right\|_2 \tag{D.65}$$

where $g$ is a standard normal independent of the other variables in the problem. Similarly

$$f_{\boldsymbol{\theta}_0}(\boldsymbol{x}) - \int_{\mathcal{M}} f_{\boldsymbol{\theta}_0}(\boldsymbol{x}')\mathrm{d}\mu^\infty(\boldsymbol{x}') = \boldsymbol{W}^{L+1}\left(\boldsymbol{\alpha}^L(\boldsymbol{x}) - \int_{\mathcal{M}}\boldsymbol{\alpha}^L(\boldsymbol{x}')\mathrm{d}\mu^\infty(\boldsymbol{x}')\right)$$

$$\overset{d}{=} g' \left\|\boldsymbol{\alpha}^L(\boldsymbol{x}) - \int_{\mathcal{M}}\boldsymbol{\alpha}^L(\boldsymbol{x}')\mathrm{d}\mu^\infty(\boldsymbol{x}')\right\|_2, \tag{D.66}$$

where $g'$ is also standard normal. With respect to the randomness of $\boldsymbol{W}^{L+1}$, these two objects are Gaussian processes with variances $\left\|\boldsymbol{\alpha}^L(\boldsymbol{x})\right\|_2^2$ and $\left\|\boldsymbol{\alpha}^L(\boldsymbol{x}) - \int_{\mathcal{M}}\boldsymbol{\alpha}^L(\boldsymbol{x}')\mathrm{d}\mu^\infty(\boldsymbol{x}')\right\|_2^2$ respectively. We next note that

$$\left\|\boldsymbol{\alpha}^L(\boldsymbol{x}) - \int_{\mathcal{M}}\boldsymbol{\alpha}^L(\boldsymbol{x}')\mathrm{d}\mu^\infty(\boldsymbol{x}')\right\|_2^2 = \left\|\int_{\mathcal{M}}\boldsymbol{\alpha}^L(\boldsymbol{x}) - \boldsymbol{\alpha}^L(\boldsymbol{x}')\mathrm{d}\mu^\infty(\boldsymbol{x}')\right\|_2^2$$

$$\leq \int_{\mathcal{M}}\left\|\boldsymbol{\alpha}^L(\boldsymbol{x}) - \boldsymbol{\alpha}^L(\boldsymbol{x}')\right\|_2^2\mathrm{d}\mu^\infty(\boldsymbol{x}')$$

$$= \int_{\mathcal{M}}\left\|\boldsymbol{\alpha}^L(\boldsymbol{x})\right\|_2^2 + \left\|\boldsymbol{\alpha}^L(\boldsymbol{x}')\right\|_2^2 - 2\left\langle\boldsymbol{\alpha}^L(\boldsymbol{x}), \boldsymbol{\alpha}^L(\boldsymbol{x}')\right\rangle\mathrm{d}\mu^\infty(\boldsymbol{x}')$$

$$\leq \sup_{\boldsymbol{x}\in\mathcal{M}}\left|\left\|\boldsymbol{\alpha}^L(\boldsymbol{x})\right\|_2^2 + \left\|\boldsymbol{\alpha}^L(\boldsymbol{x}')\right\|_2^2 - 2\left\langle\boldsymbol{\alpha}^L(\boldsymbol{x}), \boldsymbol{\alpha}^L(\boldsymbol{x}')\right\rangle\right|$$

$$\leq \sup_{\boldsymbol{x}\in\mathcal{M}}\left(\left|\begin{array}{c}\left\|\boldsymbol{\alpha}^L(\boldsymbol{x})\right\|_2^2 \\ +\left\|\boldsymbol{\alpha}^L(\boldsymbol{x}')\right\|_2^2 - 2\left\langle\boldsymbol{\alpha}^L(\boldsymbol{x}), \boldsymbol{\alpha}^L(\boldsymbol{x}')\right\rangle \\ -(2 - 2\cos\varphi^{(L)}(\nu(\boldsymbol{x}, \boldsymbol{x}'))) \\ +\left|2 - 2\cos\varphi^{(L)}(\nu(\boldsymbol{x}, \boldsymbol{x}'))\right|\end{array}\right|\right)$$

where the first inequality comes from an application of Jensen's inequality. Assuming $n, d$ satisfy the requirements of lemma D.10 and denote the event defined in it by $\mathcal{G}$. On $\mathcal{G}$, angles between features concentrate uniformly around a simple function of the angle evolution function $\varphi$, in the sense that, for all $\boldsymbol{x} \in \mathcal{M}$ simultaneously,

$$\left|\left\|\boldsymbol{\alpha}^L(\boldsymbol{x})\right\|_2^2 + \left\|\boldsymbol{\alpha}^L(\boldsymbol{x}')\right\|_2^2 - 2\left\langle\boldsymbol{\alpha}^L(\boldsymbol{x}), \boldsymbol{\alpha}^L(\boldsymbol{x}')\right\rangle - (2 - 2\cos\varphi^{(L)}(\nu(\boldsymbol{x}, \boldsymbol{x}')))\right|$$

$$\leq \left|\left\|\boldsymbol{\alpha}^L(\boldsymbol{x})\right\|_2^2 - 1\right| + \left|\left\|\boldsymbol{\alpha}^L(\boldsymbol{x}')\right\|_2^2 - 1\right| + 2\left|\left\langle\boldsymbol{\alpha}^L(\boldsymbol{x}), \boldsymbol{\alpha}^L(\boldsymbol{x}')\right\rangle - \cos\varphi^{(L)}(\nu(\boldsymbol{x}, \boldsymbol{x}'))\right| \tag{D.67}$$

$$\leq C'\sqrt{\frac{d^3 L}{n}}$$

for some constant $C'$. From lemma C.12, there exists a constant $c_0 > 0$ such that for all $\nu \in [0, \pi]$,

$$0 \leq \varphi^{(L)}(\nu) \leq \frac{1}{c_0 L}.$$

Using $\cos x \geq 1 - x^2/2$ and the above bound gives

$$1 \geq \cos\varphi^{(L)}(\nu) \geq 1 - \frac{(\varphi^{(L)}(\nu))^2}{2} \geq 1 - \frac{1}{2c_0^2 L^2}$$

and thus

$$\left|2 - 2\cos\varphi^{(L)}(\nu(\boldsymbol{x}, \boldsymbol{x}'))\right| \leq \frac{1}{c_0^2 L^2}.$$

Combining the above bound with D.67 and recalling the probability of $\mathcal{G}$ holding, we have

$$\mathbb{P}\left[\sup_{\boldsymbol{x}\in\mathcal{M}}\left|\left\|\boldsymbol{\alpha}^L(\boldsymbol{x})\right\|_2^2+\left\|\boldsymbol{\alpha}^L(\boldsymbol{x}')\right\|_2^2-2\left\langle\boldsymbol{\alpha}^L(\boldsymbol{x}),\boldsymbol{\alpha}^L(\boldsymbol{x}')\right\rangle\right|>\frac{1}{c_0^2L^2}+C'\sqrt{\frac{d^3L}{n}}\right]\le e^{-cd}. \tag{D.68}$$

On the same event we have

$$\mathbb{P}\left[\sup_{\boldsymbol{x}\in\mathcal{M}}\left\|\boldsymbol{\alpha}^L(\boldsymbol{x})\right\|_2^2>2\right]\le e^{-cd}.$$

Thus on $\mathcal{G}$ the variances of the Gaussian processes (D.65), (D.66) are uniformly bounded by 2 and $\frac{1}{c_0^2L^2}+C'\sqrt{\frac{d^3L}{n}}$ respectively. Writing for concision in the subsequent expression

$$\mathcal{E}_*\left\{\left|f_{\boldsymbol{\theta}_0}(\overline{\boldsymbol{x}})-\int_{\mathcal{M}}f_{\boldsymbol{\theta}_0}(\boldsymbol{x}')\mathrm{d}\mu^\infty(\boldsymbol{x}')\right|\le\frac{\sqrt{d}}{2}\sqrt{\frac{1}{L^2}+\sqrt{\frac{d^3L}{n}}}\right\}$$

taking a union bound over all points on the net $N_{n^{-3}n_0^{-1/2}}$ gives

$$\mathbb{P}\left[\bigcap_{\overline{\boldsymbol{x}}\in N_{n^{-3}n_0^{-1/2}}}\left\{|f_{\boldsymbol{\theta}_0}(\overline{\boldsymbol{x}})|\le\frac{\sqrt{d}}{2}\right\}\cap\mathcal{E}_*\ \middle|\ \mathcal{G}\right] \tag{D.69}$$

$$\ge1-\left|N_{n^{-3}n_0^{-1/2}}\right|e^{-cd}\ge1-e^{-c'd},$$

for some constants, since $d$ was chosen to satisfy the conditions of lemma D.10.

In addition, we see from (D.63) that on the same event, for every $\boldsymbol{x}\in\mathcal{M}$ there exists $\overline{\boldsymbol{x}}\in N_{n^{-3}n_0^{-1/2}}$ such that

$$\begin{aligned}|f_{\boldsymbol{\theta}_0}(\boldsymbol{x})-f_{\boldsymbol{\theta}_0}(\overline{\boldsymbol{x}})|&=\left|f_{\boldsymbol{\theta}_0}(\boldsymbol{x})-\int_{\mathcal{M}}f_{\boldsymbol{\theta}_0}(\boldsymbol{x}')\mathrm{d}\mu^\infty(\boldsymbol{x}')-f_{\boldsymbol{\theta}_0}(\overline{\boldsymbol{x}})-\int_{\mathcal{M}}f_{\boldsymbol{\theta}_0}(\boldsymbol{x}')\mathrm{d}\mu^\infty(\boldsymbol{x}')\right|\\&=\left|\boldsymbol{W}^{L+1}\left(\boldsymbol{\alpha}^L(\boldsymbol{x})-\boldsymbol{\alpha}^L(\overline{\boldsymbol{x}})\right)\right|\\&\le\left\|\boldsymbol{W}^{L+1}\right\|_2\left\|\boldsymbol{\alpha}^L(\boldsymbol{x})-\boldsymbol{\alpha}^L(\overline{\boldsymbol{x}})\right\|_2\le\frac{C\left\|\boldsymbol{W}^{L+1}\right\|_2}{n^{5/2}}.\end{aligned}$$

Bernstein's inequality also gives $\mathbb{P}\left[\left\|\boldsymbol{W}^{L+1}\right\|_2>C\sqrt{n}\right]\le e^{-cn}$ for some constants. Denoting the complement of this event by $\mathcal{E}$ we have that on $\mathcal{E}\cap\mathcal{G}$, for every $\boldsymbol{x}\in\mathcal{M}$ there exists $\overline{\boldsymbol{x}}\in N_{n^{-3}n_0^{-1/2}}$ such that

$$\|f_{\boldsymbol{\theta}_0}(\boldsymbol{x})-f_{\boldsymbol{\theta}_0}(\overline{\boldsymbol{x}})\|_2\le\frac{C'}{n^2}\le\frac{\sqrt{d}}{2}$$

$$\left\|f_{\boldsymbol{\theta}_0}(\boldsymbol{x})-\int_{\mathcal{M}}f_{\boldsymbol{\theta}_0}(\boldsymbol{x}')\mathrm{d}\mu^\infty(\boldsymbol{x}')-f_{\boldsymbol{\theta}_0}(\overline{\boldsymbol{x}})-\int_{\mathcal{M}}f_{\boldsymbol{\theta}_0}(\boldsymbol{x}')\mathrm{d}\mu^\infty(\boldsymbol{x}')\right\|_2\le\frac{C'}{n^2}\le\frac{\sqrt{d}}{2}\sqrt{\frac{1}{L^2}+\sqrt{\frac{d^3L}{n}}}$$

where we assumed $d\ge1,n\ge K\sqrt{L}$ for some $K$. Combining the above bound with (D.69) and taking a union bound over the complements of $\mathcal{E},\mathcal{G}$ gives

$$\mathbb{P}\left[\bigcap_{\boldsymbol{x}\in\mathcal{M}}\left\{|f_{\boldsymbol{\theta}_0}(\boldsymbol{x})|\le\sqrt{d}\right\}\cap\left\{\left|f_{\boldsymbol{\theta}_0}(\boldsymbol{x})-\int_{\mathcal{M}}f_{\boldsymbol{\theta}_0}(\boldsymbol{x}')\mathrm{d}\mu^\infty(\boldsymbol{x}')\right|\le\sqrt{\frac{d}{L^2}+d^{5/2}\sqrt{\frac{L}{n}}}\right\}\right]$$

$$\ge1-e^{-cd}-\mathbb{P}\left[\mathcal{G}^c\right]-\mathbb{P}\left[\mathcal{E}^c\right]$$

$$\ge1-e^{-c'd}-e^{-c''n}\ge1-e^{-c'''d}.$$

From the first result, we also obtain that with the the same probability $\|\zeta\|_{L^\infty}\le1+\sqrt{d}$. By worsening the constant in the tail we can simplify this to $\|\zeta\|_{L^\infty}\le\sqrt{d}$.

Defining

$$\hat{\zeta}(\boldsymbol{x}) = -f_\star(\boldsymbol{x}) + \int_{\mathcal{M}} f_{\boldsymbol{\theta}_0}(\boldsymbol{x}') \mathrm{d}\mu^\infty(\boldsymbol{x}'),$$

since $\hat{\zeta}(\boldsymbol{x}) - \zeta(\boldsymbol{x}) = f_{\boldsymbol{\theta}_0}(\boldsymbol{x}) - \int_{\mathcal{M}} f_{\boldsymbol{\theta}_0}(\boldsymbol{x}') \mathrm{d}\mu^\infty(\boldsymbol{x}')$, it follows that

$$\mathbb{P}\left[\left\|\hat{\zeta} - \zeta\right\|_{L^\infty} \le \sqrt{\frac{d}{L^2} + d^{5/2}\sqrt{\frac{L}{n}}}\right] \ge 1 - e^{-c'''d}.$$

$\square$

**Lemma D.12.** *For some integer $d_0$, assume $n, L, d$ satisfy the requirements of lemmas D.8 and D.14, meaning that there exist absolute constants $K, K', K'', K''', K'''' > 0$ such that for any $d \ge \max\{Kd_0 \log(nn_0 C_{\mathcal{M}}), K' \log L\}$, if $n \ge \max\{K''d^4 L, K'''L \log n, K''''\}$ then*

1. *If $d_0 = 1$ and $n \ge K'''''\max\left\{d^2 L, \left(\frac{\kappa}{c_\tau}\right)^{1/3}, \kappa^{2/5}\right\}$ where $K'''''$ is some absolute constant. $\kappa$ and $c_\lambda$ are the extrinsic curvature and injectivity coefficient defined in Section 2.1, then on an event of probability at least $1 - e^{-cd}$, one has*

$$\left\|f_{\boldsymbol{\theta}_0}\big|_{\mathcal{M}_\pm}\right\|_{\mathrm{Lip}} \le \sqrt{d}$$

*for a numerical constant $c$, and where the Lipschitz seminorm is taken with respect to the Riemannian distance on $\mathcal{M}_\pm$.*

2. *If $\mathcal{M} = \mathbb{S}^{n_0-1}$ so that $d_0 = n_0 - 1$, then on an event of probability at least $1 - e^{-cd}$, one has*

$$\left\|f_{\boldsymbol{\theta}_0}\big|_{\mathbb{S}^{n_0-1}}\right\|_{\mathrm{Lip}} \le \sqrt{d}$$

*for a numerical constant $c$.*

*Proof.* We recall

$$f_{\boldsymbol{\theta}_0}(\boldsymbol{x}) = \boldsymbol{W}^{L+1}\boldsymbol{\alpha}^L(\boldsymbol{x}).$$

Let $\mathcal{M}_\star$ denote a connected component of $\mathcal{M}$. Let $\boldsymbol{x}_1, \boldsymbol{x}_2 \in \mathcal{M}_\star$, and fix a smooth unit-speed geodesic $\gamma : [0, \mathrm{dist}_{\mathcal{M}_\star}(\boldsymbol{x}_1, \boldsymbol{x}_2)] \to \mathcal{M}_\star$ such that $\gamma(0) = \boldsymbol{x}_1$ and $\gamma(\mathrm{dist}_{\mathcal{M}_\star}(\boldsymbol{x}_1, \boldsymbol{x}_2)) = \boldsymbol{x}_2$. The absolute continuity of $f_{\boldsymbol{\theta}_0}\big|_{\mathcal{M}_\pm} \circ \gamma$ follows from an argument almost identical to the one employed in the proof of Lemma B.8, and we obtain in particular the bound

$$|f_{\boldsymbol{\theta}_0}(\boldsymbol{x}_1) - f_{\boldsymbol{\theta}_0}(\boldsymbol{x}_2)| = \left|\int_0^{\mathrm{dist}_{\mathcal{M}_\star}(\boldsymbol{x}_1, \boldsymbol{x}_2)} \left\langle \gamma'(t), \left(\boldsymbol{W}^1\right)^* \boldsymbol{\beta}^0(\gamma(t)) \right\rangle \mathrm{d}t\right|.$$

Because $\gamma$ is a unit-speed geodesic, we have for all $t$

$$\boldsymbol{P}_{T_{\gamma(t)}\mathcal{M}_\star}\gamma'(t) = (\gamma'(t)\gamma'(t)^*)\,\gamma'(t) = \gamma'(t),$$

and so in particular, by the triangle inequality and Cauchy-Schwarz

$$|f_{\boldsymbol{\theta}_0}(\boldsymbol{x}_1) - f_{\boldsymbol{\theta}_0}(\boldsymbol{x}_2)| \le \mathrm{dist}_{\mathcal{M}_\star}(\boldsymbol{x}_1, \boldsymbol{x}_2) \sup_{\boldsymbol{x}\in\mathcal{M}_\star} \left\|\boldsymbol{P}_{T_{\boldsymbol{x}}\mathcal{M}_\star}\left(\boldsymbol{W}^1\right)^* \boldsymbol{\beta}^0(\boldsymbol{x})\right\|_2. \tag{D.70}$$

This implies

$$\left\|f_{\boldsymbol{\theta}_0}\big|_{\mathcal{M}_\star}\right\|_{\mathrm{Lip}} \le \sup_{\boldsymbol{x}\in\mathcal{M}_\star} \left\|\boldsymbol{P}_{T_{\boldsymbol{x}}\mathcal{M}_\star}\boldsymbol{W}^{1*}\boldsymbol{\beta}^0(\boldsymbol{x})\right\|_2.$$

We next write

$$\boldsymbol{W}^1 = \boldsymbol{W}^1\boldsymbol{x}\boldsymbol{x}^* + \boldsymbol{W}^1\left(\boldsymbol{I} - \boldsymbol{x}\boldsymbol{x}^*\right) = \boldsymbol{G}^1 + \boldsymbol{H}^1.$$

$T_{\boldsymbol{x}}\mathcal{M}_\star$ can be identified with a linear subspace of $\mathbb{R}^{n_0}$ of dimension $d_0$. Since it is also a subspace of $T_{\boldsymbol{x}}\mathbb{S}^{n_0-1}$, $\boldsymbol{P}_{T_{\boldsymbol{x}}\mathcal{M}_\star}\boldsymbol{x} = 0$ and hence

$$\boldsymbol{P}_{T_{\boldsymbol{x}}\mathcal{M}_\star}\boldsymbol{W}^{1*} = \boldsymbol{P}_{T_{\boldsymbol{x}}\mathcal{M}_\star}\boldsymbol{H}^{1*} \overset{d}{=} \boldsymbol{P}_{T_{\boldsymbol{x}}\mathcal{M}_\star}\left(\boldsymbol{I} - \boldsymbol{x}\boldsymbol{x}^*\right)\tilde{\boldsymbol{W}}^{1*} = \boldsymbol{P}_{T_{\boldsymbol{x}}\mathcal{M}_\star}\tilde{\boldsymbol{W}}^{1*}$$

where $\tilde{\boldsymbol{W}}^{1*}$ is a copy of $\boldsymbol{W}^{1*}$ that is independent of all the other variables in the problem (since $\boldsymbol{\beta}^0(\boldsymbol{x})$ depends only on $\boldsymbol{G}^1$).

We first consider the case $d_0 = n_0 - 1$, and subsequently the case $d_0 = 1$. We note that

$$\left\|\boldsymbol{P}_{T_{\boldsymbol{x}}\mathcal{M}_\star}\tilde{\boldsymbol{W}}^{1*}\boldsymbol{\beta}^0(\boldsymbol{x})\right\|_2^2 \le \left\|\tilde{\boldsymbol{W}}^{1*}\boldsymbol{\beta}^0(\boldsymbol{x})\right\|_2^2 = \sum_{i=1}^{n_0}\left(\tilde{\boldsymbol{W}}_i^{1*}\boldsymbol{\beta}^0(\boldsymbol{x})\right)^2, \tag{D.71}$$

Considering a single summand in the above expression, repeated application of the rotational invariance of the gaussian distribution gives

$$\left(\tilde{\boldsymbol{W}}_i^{1*}\boldsymbol{\beta}^0(\boldsymbol{x})\right)^2 \qquad \overset{d}{=} \quad \frac{2\left\|\boldsymbol{\beta}^0(\boldsymbol{x})\right\|_2^2}{n}g_{i,\mathcal{I}(\boldsymbol{x})}^2$$

$$= \quad \frac{2}{n}\left\|\boldsymbol{W}^{L+1}\boldsymbol{\Gamma}^{L:2}(\boldsymbol{x})\boldsymbol{P}_{I_1(\boldsymbol{x})}\right\|_2^2 g_{i,\mathcal{I}(\boldsymbol{x})}^2$$

$$\le \quad \frac{2}{n}\left\|\boldsymbol{W}^{L+1}\boldsymbol{\Gamma}^{L:2}(\boldsymbol{x})\right\|_2^2 g_{i,\mathcal{I}(\boldsymbol{x})}^2$$

$$\overset{d}{=} \quad \frac{2}{n}\left\|\boldsymbol{\Gamma}^{L:2}(\boldsymbol{x})\boldsymbol{W}^{L+1}\right\|_2^2 g_{i,\mathcal{I}(\boldsymbol{x})}^2$$

$$\overset{d}{=} \quad \frac{2}{n}\left\|\boldsymbol{\Gamma}^{L:2}(\boldsymbol{x})\boldsymbol{e}_1\right\|_2^2\left\|\boldsymbol{W}^{L+1}\right\|_2^2 g_{i,\mathcal{I}(\boldsymbol{x})}^2 \tag{D.72}$$

where $g_{i,\mathcal{I}(\boldsymbol{x})}$ is a standard normal variable that depends on $i$ and the support patterns $\mathcal{I}(\boldsymbol{x}) = \{I_1(\boldsymbol{x}),\ldots,I_L(\boldsymbol{x})\}$, since $\boldsymbol{\beta}^0(\boldsymbol{x})$ depends on $\boldsymbol{x}$ only through $\mathcal{I}(\boldsymbol{x})$ Similarly, the dependence on $\boldsymbol{x}$ in the first factor in (D.72) is only through $\mathcal{I}(\boldsymbol{x})$. We now show how to control such terms uniformly on $\mathcal{M}_\star$.

Define a net $N_{n^{-3}n_0^{-1/2}}$ as in Appendix D.3.1. According to Lemma C.4, $\left|N_{n^{-3}n_0^{-1/2}}\right| \le e^{3\log(C_{\mathcal{M}_\star}nn_0)d_0}$. Assume $n, L, d$ satisfy the requirements of Lemma D.8 and denote this event defined in that lemma by $\mathcal{E}$. We also define sets of support patterns

$$\overline{\mathcal{J}}(\overline{\boldsymbol{x}}) = \left\{\mathcal{J} = \{J_1,\ldots,J_L\}\Big|\sum_{\ell=1}^L|J_\ell \ominus I_\ell(\overline{\boldsymbol{x}})| \le d\right\}, \tag{D.73}$$

$$\overline{\mathcal{J}}(N_{n^{-3}n_0^{-1/2}}) = \bigcup_{\overline{\boldsymbol{x}}\in N_{n^{-3}n_0^{-1/2}}}\overline{\mathcal{J}}(\overline{\boldsymbol{x}}).$$

On $\mathcal{E}$, $\bigcup_{\boldsymbol{x}\in\mathcal{M}_\star}\{\mathcal{I}(\boldsymbol{x})\} \subseteq \overline{\mathcal{J}}(N_{n^{-3}n_0^{-1/2}})$, and additionally for any $\boldsymbol{x} \in \mathcal{M}_\star$, then there exists some $\overline{\boldsymbol{x}} \in N_{n^{-3}n_0^{-1/2}}$ such that $\boldsymbol{x} \in \mathcal{N}_{n^{-3}n_0^{-1/2}}(\overline{\boldsymbol{x}})$ and $\mathcal{I}(\boldsymbol{x}) \in \overline{\mathcal{J}}(\overline{\boldsymbol{x}})$. We now show that on $\mathcal{E}$, the supports $\mathcal{I}(\boldsymbol{x})$ satisfy the requirements of Lemma D.14 with $\delta_s = d, K_s = Cn^{-5/2}$, with the anchor point in the statement of that lemma chosen to be $\overline{\boldsymbol{x}}$. The value of $\delta_s$ is satisfied by the supports at every point on the manifold on $\mathcal{E}$ from the definition of this event. From the definition of the stable sign consistency property (SSC) in Appendices D.3.1 and D.3.2, the only features whose sign can differ between $\overline{\boldsymbol{x}}$ to $\boldsymbol{x}$ are the risky features, and from the bound on their norm in the definition of $\mathsf{SSC}(L, n^{-3}n_0^{-1/2}, Cn^{-3})$ we obtain for all $\ell$

$$\left\|\left(\boldsymbol{P}_{J_\ell} - \boldsymbol{P}_{I_\ell(\overline{\boldsymbol{x}})}\right)\boldsymbol{\rho}^\ell(\overline{\boldsymbol{x}})\right\|_{L^\infty} \le \frac{C}{n^3} \Rightarrow \left\|\left(\boldsymbol{P}_{J_\ell} - \boldsymbol{P}_{I_\ell(\overline{\boldsymbol{x}})}\right)\boldsymbol{\rho}^\ell(\overline{\boldsymbol{x}})\right\|_2 \le \frac{C}{n^{5/2}}$$

where in the last inequality we used Lemma G.10. It follows that if $n \ge KLd$ for some $K$, the requirements of Lemma D.14 are satisfied if we make the choice $\mathcal{E} = \mathcal{E}_{\delta K}$.

We would next like to apply Lemma D.14 for every possible support pattern in $\overline{\mathcal{J}}(N_{n^{-3}n_0^{-1/2}})$, which requires that we first bound the cardinality of this set. Note that $\overline{\mathcal{J}}(\overline{\boldsymbol{x}})$ is the . Thus

$$|\overline{\mathcal{J}}(\overline{\boldsymbol{x}})| \le \sum_{i=0}^{\lfloor d\rfloor}\binom{Ln}{i} \le \lfloor d\rfloor\left(\frac{eLn}{\lfloor d\rfloor}\right)^{\lfloor d\rfloor} \le (Ln)^{Cd}$$

for some $C$. Using the bound on the cardinality of the net from Lemma C.4, the size of this set can be bounded, giving

$$\left|\overline{\mathcal{J}}(N_{n^{-3}n_0^{-1/2}})\right| \leq \left|N_{n^{-3}n_0^{-1/2}}\right| \sum_{i=0}^{\lceil d \rceil} \binom{n}{i} \leq e^{3\log(C_{\mathcal{M}_\star}nn_0)d_0+C\log(Ln)d}. \tag{D.74}$$

We can now apply Lemma D.14 with $\mathcal{E} = \mathcal{E}_{\delta K}$ to bound $\left\|\Gamma^{\ell:2}(\boldsymbol{x})\boldsymbol{e}_1\right\|_2^2$ for all $\ell$ on $\mathcal{E}$, taking a union bound over all possible supports. Bernstein's inequality and an exponential tail bound can be used to bound the second factor and third factors in (D.72) respectively. Using (D.74) to bound the number of supports we need to uniformize over (since on $\mathcal{E}$, $\bigcup_{\boldsymbol{x}\in\mathcal{M}_\star}\{\mathcal{I}(\boldsymbol{x})\} \subseteq \overline{\mathcal{J}}(N_{n^{-3}n_0^{-1/2}})$) and the bound on the size of the net in Lemma C.4 and Appendix D.3.1, we obtain

$$
\begin{aligned}
&\mathbb{P}\left[\forall \boldsymbol{x} \in \mathcal{M}_\star : \frac{2}{n}\mathbb{1}_{\mathcal{E}}\left\|\Gamma^{L:2}(\boldsymbol{x})\boldsymbol{e}_1\right\|_2^2 \left\|\boldsymbol{W}^{L+1}\right\|_2^2 g_{i,\mathcal{I}(\boldsymbol{x})}^2 \leq d\right] \\
=&\mathbb{P}\left[\begin{array}{l}\forall \overline{\boldsymbol{x}} \in N_{n^{-3}n_0^{-1/2}}, \\ \forall \boldsymbol{x} \in \mathcal{N}_{n^{-3}n_0^{-1/2}}(\overline{\boldsymbol{x}})\end{array} : \frac{2}{n}\mathbb{1}_{\mathcal{E}}\left\|\Gamma^{L:2}(\boldsymbol{x})\boldsymbol{e}_1\right\|_2^2 \left\|\boldsymbol{W}^{L+1}\right\|_2^2 g_{i,\mathcal{I}(\boldsymbol{x})}^2 \leq d\right] \\
\geq& 1 - \left|N_{n^{-3}n_0^{-1/2}}\right| e^{C\log(n)d}e^{-c\frac{n}{L}} - \left|N_{n^{-3}n_0^{-1/2}}\right| e^{C\log(n)d}e^{-c'd} - e^{-c''n} \\
\geq& 1 - e^{3\log(C_{\mathcal{M}_\star}nn_0)d_0+C\log(n)d-c\frac{n}{L}} - e^{3\log(C_{\mathcal{M}_\star}nn_0)d_0+C\log(n)d-c'd} - e^{-c''n} \\
\geq& 1 - e^{-c'''d}
\end{aligned}
\tag{D.75}
$$

where we assume $d \geq K\log(C_{\mathcal{M}_\star}nn_0)d_0, n \geq K'Ld^2$ for some $K, K'$. Since according to Lemma D.8 the event $\mathcal{E}$ holds with probability greater than $1 - e^{cd}$ for some $c$, we can remove the indicator in the bound above by assuming $d \geq K$ for some absolute constant $K$ and worsening the constant in the bound.

We can now complete the proof for $d_0 = n_0 - 1$. Since we are interested in bounding the sum (D.71) uniformly, we can bound $\sum_{i=1}^{n_0} g_{i,\mathcal{I}(\boldsymbol{x})}^2$ as well using Bernstein's inequality and uniformizing as above obtain

$$
\begin{aligned}
&\mathbb{P}\left[\forall \boldsymbol{x} \in \mathcal{M}_\star : \frac{2}{n}\mathbb{1}_{\mathcal{E}}\left\|\Gamma^{L:2}(\boldsymbol{x})\boldsymbol{e}_1\right\|_2^2 \left\|\boldsymbol{W}^{L+1}\right\|_2^2 \sum_{i=1}^{n_0} g_{i,\mathcal{I}(\boldsymbol{x})}^2 \leq C(n_0+d)\right] \\
=&\mathbb{P}\left[\begin{array}{l}\forall \overline{\boldsymbol{x}} \in N_{n^{-3}n_0^{-1/2}}, \\ \forall \boldsymbol{x} \in \mathcal{N}_{n^{-3}n_0^{-1/2}}(\overline{\boldsymbol{x}})\end{array} : \frac{2}{n}\mathbb{1}_{\mathcal{E}}\left\|\Gamma^{L:2}(\boldsymbol{x})\boldsymbol{e}_1\right\|_2^2 \left\|\boldsymbol{W}^{L+1}\right\|_2^2 \sum_{i=1}^{n_0} g_{i,\mathcal{I}(\boldsymbol{x})}^2 \leq C(n_0+d)\right] \\
\geq& 1 - \left|N_{n^{-3}n_0^{-1/2}}\right| e^{C\log(n)d}e^{-c\frac{n}{L}} - \left|N_{n^{-3}n_0^{-1/2}}\right| e^{C\log(n)d}e^{-c'd} - e^{-c''n} \\
\geq& 1 - e^{3\log(C_{\mathcal{M}_\star}nn_0)d_0+C\log(n)d-c\frac{n}{L}} - e^{3\log(C_{\mathcal{M}_\star}nn_0)d_0+C\log(n)d-c'd} - e^{-c''n} \\
\geq& 1 - e^{-c'''d}
\end{aligned}
$$

where we assume $d \geq K\log(C_{\mathcal{M}_\star}nn_0)d_0, n \geq K'Ld^2$ for some $K, K'$. As above, we can remove the indicator in the bound above by assuming $d \geq K$ for some absolute constant $K$ and worsening the constant in the bound. Worsening constants in the failure probability, we can replace the residual in the above expression by $d$. Using (D.70), we obtain that with the same probability the Lipschitz constant of $f_{\boldsymbol{\theta}_0}$ on $\mathbb{S}^{n_0-1}$ is bounded by $\sqrt{d}$.

We now consider $d_0 = 1$. Recall that for any $\boldsymbol{x} \in \mathcal{M}_\star$, then there exists some $\overline{\boldsymbol{x}} \in N_{n^{-3}n_0^{-1/2}}$ such that $\boldsymbol{x} \in \mathcal{N}_{n^{-3}n_0^{-1/2}}(\overline{\boldsymbol{x}})$ where $N_{n^{-3}n_0^{-1/2}}$ is the net defined earlier. The gradient vector at $\boldsymbol{x}$ takes the form

$$\boldsymbol{P}_{T_{\boldsymbol{x}}\mathcal{M}}\tilde{\boldsymbol{W}}^{1*}\boldsymbol{\beta}^0(\boldsymbol{x}) = \boldsymbol{P}_{T_{\overline{\boldsymbol{x}}}\mathcal{M}}\tilde{\boldsymbol{W}}^{1*}\boldsymbol{\beta}^0(\boldsymbol{x}) + \left(\boldsymbol{P}_{T_{\boldsymbol{x}}\mathcal{M}} - \boldsymbol{P}_{T_{\overline{\boldsymbol{x}}}\mathcal{M}}\right)\tilde{\boldsymbol{W}}^{1*}\boldsymbol{\beta}^0(\boldsymbol{x}). \tag{D.76}$$

We proceed to bound the first term in the above equation uniformly over $\mathcal{M}_\star$. Since in the $d_0 = 1$ case $T_{\overline{\boldsymbol{x}}}\mathcal{M}_\star$ can be identified with a linear subspace of $\mathbb{R}^{n_0}$ of dimension 1, we can write the

projection operator as

$$\boldsymbol{P}_{T_{\overline{\boldsymbol{x}}}\mathcal{M}_\star} = \boldsymbol{v}_{\overline{\boldsymbol{x}}}\boldsymbol{v}_{\overline{\boldsymbol{x}}}^* \tag{D.77}$$

for some unit norm $\boldsymbol{v}_{\overline{\boldsymbol{x}}}$. We then obtain from rotational invariance of the gaussian distribution that

$$\left\|\boldsymbol{P}_{T_{\overline{\boldsymbol{x}}}\mathcal{M}}\tilde{\boldsymbol{W}}^{1*}\boldsymbol{\beta}^0(\boldsymbol{x})\right\|_2^2 = \left\|\boldsymbol{v}_{\overline{\boldsymbol{x}}}\boldsymbol{v}_{\overline{\boldsymbol{x}}}^*\tilde{\boldsymbol{W}}^{1*}\boldsymbol{\beta}^0(\boldsymbol{x})\right\|_2^2$$

$$\overset{d}{=} \frac{2\left\|\boldsymbol{\beta}^0(\boldsymbol{x})\right\|_2^2}{n}g_{\overline{\boldsymbol{x}}}^2$$

$$\leq \frac{2}{n}\left\|\boldsymbol{\Gamma}^{L:2}(\boldsymbol{x})\boldsymbol{e}_1\right\|_2^2\left\|\boldsymbol{W}^{L+1}\right\|_2^2 g_{\overline{\boldsymbol{x}}}^2,$$

where $g_{\overline{\boldsymbol{x}}}$ is a standard normal variable and the last bound comes from a calculation identical to (D.72). Under the same assumptions on $d, L, n$ as before, we can bound this expression uniformly using (D.75), and additionally take a union bound over the net to account for all possible choices of $\overline{\boldsymbol{x}}$. This gives

$$\mathbb{P}\left[\begin{matrix}\forall\overline{\boldsymbol{x}} \in N_{n^{-3}n_0^{-1/2}}, \\ \forall\boldsymbol{x} \in \mathcal{N}_{n^{-3}n_0^{-1/2}}(\overline{\boldsymbol{x}})\end{matrix} : \frac{2}{n}\mathbb{1}_{\mathcal{E}}\left\|\boldsymbol{\Gamma}^{L:2}(\boldsymbol{x})\boldsymbol{e}_1\right\|_2^2\left\|\boldsymbol{W}^{L+1}\right\|_2^2 g_{\overline{\boldsymbol{x}}}^2 \leq d\right]$$

$$\geq 1 - \left|N_{n^{-3}n_0^{-1/2}}\right|e^{-c'd}$$

$$\geq 1 - e^{3\log(C_{\mathcal{M}_\star}nn_0)d_0 - c'd}$$

$$\geq 1 - e^{-c''d}$$

where we assume $d \geq K\log(C_{\mathcal{M}_\star}nn_0)$ for some $K$. This gives control of the first term in (D.76) uniformly on $\mathcal{M}$.

We now turn to controlling the second term in (D.76). For some $\boldsymbol{x}$, choose $\overline{\boldsymbol{x}} \in N_{n^{-3}n_0^{-1/2}}$ such that $\boldsymbol{x} \in \mathcal{N}_{n^{-3}n_0^{-1/2}}(\overline{\boldsymbol{x}})$. Define a unit-speed curve $\boldsymbol{\gamma} : [0, s] \to \mathcal{M}$ such that $\boldsymbol{\gamma}(0) = \overline{\boldsymbol{x}}, \boldsymbol{\gamma}(s) = \boldsymbol{x}$. Since the curvature of $\mathcal{M}$ is bounded by $\kappa$, we have

$$\forall s' \in [0, s] : \left\|\boldsymbol{\gamma}''(s')\right\|_2 \leq \kappa.$$

Denote by $r$ the geodesic distance between $\boldsymbol{x}$ and $\boldsymbol{x}'$. Since the euclidean distance between them is bounded by $n^{-3}n_0^{-1/2}$, assuming $n > K$ for some $K$ implies that $r < C'n^{-3}n_0^{-1/2}$ for some $C'$. If we now demand $n^3 \geq C'\frac{\kappa}{c_\lambda}$ which implies $\frac{C'}{n^3 n_0^{1/2}} \leq \frac{c_\tau}{\kappa}$, (A.2) gives

$$s \leq \frac{C}{n^3 n_0^{1/2}},$$

for some $C > 0$. For $\boldsymbol{v}^{\overline{\boldsymbol{x}}}, \boldsymbol{v}^{\boldsymbol{x}}$ defined as in (D.77) we have $\boldsymbol{\gamma}'(0) = \boldsymbol{v}^{\overline{\boldsymbol{x}}}, \boldsymbol{\gamma}'(s) = \boldsymbol{v}^{\boldsymbol{x}}$. Combining the previous two results, it follows that

$$\left\|\boldsymbol{v}^{\overline{\boldsymbol{x}}} - \boldsymbol{v}^{\boldsymbol{x}}\right\|_2 = \left\|\boldsymbol{\gamma}'(0) - \boldsymbol{\gamma}'(s)\right\|_2 = \left\|\int_{s'=0}^s \boldsymbol{\gamma}''(s')ds'\right\|_2 \leq s\kappa \leq \frac{C\kappa}{n^3 n_0^{1/2}}.$$

A straightforward calculation then gives

$$\left\|\boldsymbol{P}_{T_{\boldsymbol{x}}\mathcal{M}} - \boldsymbol{P}_{T_{\overline{\boldsymbol{x}}}\mathcal{M}}\right\| = \left\|\boldsymbol{v}^{\boldsymbol{x}}\boldsymbol{v}^{\boldsymbol{x}*} - \boldsymbol{v}^{\overline{\boldsymbol{x}}}\boldsymbol{v}^{\overline{\boldsymbol{x}}*}\right\| = \frac{1}{2}\left\|\boldsymbol{v}^{\overline{\boldsymbol{x}}} - \boldsymbol{v}^{\boldsymbol{x}}\right\|_2\left\|\boldsymbol{v}^{\overline{\boldsymbol{x}}} + \boldsymbol{v}^{\boldsymbol{x}}\right\|_2$$

$$\leq \left\|\boldsymbol{v}^{\overline{\boldsymbol{x}}} - \boldsymbol{v}^{\boldsymbol{x}}\right\|_2 \leq \frac{C\kappa}{n^3 n_0^{1/2}}.$$

If we now use Lemma D.13 to control the norms of the backward features uniformly, a standard bound on the norm of a Gaussian matrix to give $\mathbb{P}\left[\left\|\tilde{\boldsymbol{W}}^1\right\| > C\left(1 + \sqrt{\frac{n_0}{n}}\right)\right] \leq e^{-cn}$, and assume $n \geq \kappa^{2/5}$ we obtain that

$$\mathbb{P}\left[\forall\overline{\boldsymbol{x}} \in N_{n^{-3}n_0^{-1/2}}, \boldsymbol{x} \in \mathcal{N}_{n^{-3}n_0^{-1/2}}(\overline{\boldsymbol{x}}) : \left|\left(\boldsymbol{P}_{T_{\boldsymbol{x}}\mathcal{M}} - \boldsymbol{P}_{T_{\overline{\boldsymbol{x}}}\mathcal{M}}\right)\tilde{\boldsymbol{W}}^{1*}\boldsymbol{\beta}^0(\boldsymbol{x})\right| \leq C\right]$$

$$\geq 1 - e^{-cd} - e^{-c'n} \geq 1 - e^{-c''d}.$$

Combining the above with (D.75) and using (D.76) and taking a union bound over the failure probability of $\mathcal{E}$ which results in a worsening of constants completes the proof. We can additionally rescale $d$ to obtain a final bound on the Lipschitz constant of $\sqrt{d}$ instead of $C\sqrt{d}$, which also results in a worsening of constants.

$\square$

**Lemma D.13.** *There are absolute constants $c, C > 0$ and absolute constants $K_1, \ldots, K_4 > 0$ such that for any $d \geq K d_0 \log(n n_0 C_{\mathcal{M}})$, if $n \geq K' d^4 L$, then there exists an event $\mathcal{E}$ such that*

1. *On $\mathcal{E}$, we have*

$$\forall \ell \in [L], \; \left| \langle \boldsymbol{\beta}^{\ell-1}(\boldsymbol{x}), \boldsymbol{\beta}^{\ell-1}(\boldsymbol{x}') \rangle - \frac{n}{2} \prod_{\ell'=\ell-1}^{L-1} \left( 1 - \frac{\varphi^{(\ell')}(\angle(\boldsymbol{x}, \boldsymbol{x}'))}{\pi} \right) \right| \leq C\sqrt{d^4 n L}$$

   *simultaneously for every $(\boldsymbol{x}, \boldsymbol{x}') \in \mathcal{M} \times \mathcal{M}$;*

2. $\mathbb{P}[\mathcal{E}] \geq 1 - e^{-cd}$.

*Proof.* Let $\mathcal{E}_1$ denote the event studied in Lemma D.8, with $C_0$ denoting the absolute constant appearing in the $\mathsf{SSC}(L)$ condition there; choose $d \geq K d_0 \log(n n_0 C_{\mathcal{M}})$ and $n$ sufficiently large to make the measure bound applicable. We will need to apply Lemma D.23 together with a derandomization argument to prove the claim; we appeal to the same residual checks at the beginning of the proof of Lemma D.6 to see that on $\mathcal{E}_1$, the dominating residual in Lemma D.23 under the scalings of $d$ and $n$ we enforce here is of size $C\sqrt{d^4 nL}$.

For any subset $S \subset [L] \times [n]$, we write $S_\ell = \{i \in [n] \mid (\ell, i) \in S\}$, and we define $\mathcal{S}(S) = \{-1, +1\}^{|S_1|} \times \cdots \times \{-1, +1\}^{|S_L|}$ for the set of "lists" of sign patterns with sizes adapted to these projections of $S$, with the convention $\{-1, +1\}^0 = \{0\}$. If $\Sigma = \{\boldsymbol{\sigma}_1, \ldots, \boldsymbol{\sigma}_L\} \in \mathcal{S}(S)$ is such a list of sign vectors and $\Delta \geq 0$, we define

$$\tilde{I}_\ell(\boldsymbol{x}, S, \Sigma, \Delta) = \mathrm{supp} \left( \mathbb{1}_{\boldsymbol{\rho}^\ell(\boldsymbol{x}) > \sum_{i \in S_\ell} ((\boldsymbol{\sigma}_\ell)_i \Delta) \boldsymbol{e}_i} \right),$$

which is a sort of two-sided robust analogue of the support of $\boldsymbol{\alpha}^\ell(\boldsymbol{x})$: notice that when $S = \varnothing$ we have $\tilde{I}_\ell(\boldsymbol{x}, S, \Sigma, \Delta) = I_\ell(\boldsymbol{x})$. We also define for $\ell = 0, 1, \ldots, L-1$

$$\tilde{\boldsymbol{\beta}}^\ell_{S,\Sigma,\Delta}(\boldsymbol{x}) = \left( \boldsymbol{W}^{L+1} \boldsymbol{P}_{\tilde{I}_L(\boldsymbol{x},S,\Sigma,\Delta)} \boldsymbol{W}^L \boldsymbol{P}_{\tilde{I}_{L-1}(\boldsymbol{x},S,\Sigma,\Delta)} \cdots \boldsymbol{W}^{\ell+2} \boldsymbol{P}_{\tilde{I}_{\ell+1}(\boldsymbol{x},S,\Sigma,\Delta)} \right)^*,$$

a generalized backward feature induced by these robust support patterns. Writing for concision

$$\mathscr{S}_{\bar{\boldsymbol{x}}, \bar{\boldsymbol{x}}', S, S', \Sigma, \Sigma'} = \left\{ \exists \ell \in [L] : \begin{array}{c} \left| \langle \tilde{\boldsymbol{\beta}}^{\ell-1}_{S,\Sigma,C_0 n^{-3}}(\bar{\boldsymbol{x}}), \tilde{\boldsymbol{\beta}}^{\ell-1}_{S',\Sigma',C_0 n^{-3}}(\bar{\boldsymbol{x}}') \rangle \right. \\ \left. - \frac{n}{2} \prod_{\ell'=\ell-1}^{L-1} \left( 1 - \frac{\varphi^{(\ell')}(\angle(\bar{\boldsymbol{x}}, \bar{\boldsymbol{x}}'))}{\pi} \right) \right| \\ > C_1 \sqrt{d^4 nL \log^4 n} \end{array} \right\},$$

where $C_1 > 0$ is an absolute constant we will specify below to make the event hold with high probability, we then define the event[11]

$$\mathcal{E}_2 = \bigcup_{\substack{\bar{\boldsymbol{x}} \in N_{n^{-3}} \\ \bar{\boldsymbol{x}}' \in N_{n^{-3}}}} \bigcup_{\substack{S \subset [L] \times [n] \\ S' \subset [L] \times [n] \\ |S| \leq d, |S'| \leq d}} \bigcup_{\substack{\Sigma \in \mathcal{S}(S) \\ \Sigma' \in \mathcal{S}(S')}} \mathscr{S}_{\bar{\boldsymbol{x}}, \bar{\boldsymbol{x}}', S, S', \Sigma, \Sigma'},$$

There are no more than $\sum_{k=0}^{d} \binom{nL}{k} \leq n^{4d}$ ways to choose the subset $S$ in this union, and for a fixed $S$ there are no more than $2^d$ ways to choose the sign pattern $\Sigma$. Thus, there no more than

---

[11]To see that this set is indeed an event, use that $\tilde{\boldsymbol{\beta}}^\ell_{S,\Sigma,\Delta}(\boldsymbol{x})$ is a continuous function of the network weights except with respect to the support projections; but $x \mapsto \mathbb{1}_{x>0}$ is increasing, hence Borel-measurable, and so the set consists of a finite union of Borel-measurable sets.

$\exp(10d\log n + 12d_0\log(nn_0 C_{\mathcal{M}}))$ elements in the union, and under the condition on $d$ this number is no larger than $n^{11d}$. For concision, write

$$\xi_\ell(\bar{\boldsymbol{x}}, \bar{\boldsymbol{x}}') = \frac{n}{2}\prod_{\ell'=\ell}^{L-1}\left(1 - \frac{\varphi^{(\ell')}(\angle(\bar{\boldsymbol{x}}, \bar{\boldsymbol{x}}'))}{\pi}\right).$$

For any instantiation of these parameters, Lemma D.23 and a union bound give

$$\mathbb{P}\Big[\exists \ell \in [L] : \Big|\Big\langle \tilde{\boldsymbol{\beta}}^{\ell-1}_{S,\Sigma,C_0 n^{-3}}(\bar{\boldsymbol{x}}), \tilde{\boldsymbol{\beta}}^{\ell-1}_{S',\Sigma',C_0 n^{-3}}(\bar{\boldsymbol{x}}')\Big\rangle - \xi_{\ell-1}(\bar{\boldsymbol{x}}, \bar{\boldsymbol{x}}')\Big| > C\sqrt{d^4 nL}\Big]$$
$$\leq \mathbb{P}[\mathcal{E}_1^{\mathsf{c}}] + \mathbb{P}\Big[\exists \ell \in [L] : \Big|\mathbb{1}_{\mathcal{E}_1}\Big\langle \tilde{\boldsymbol{\beta}}^{\ell-1}_{S,\Sigma,C_0 n^{-3}}(\bar{\boldsymbol{x}}), \tilde{\boldsymbol{\beta}}^{\ell-1}_{S',\Sigma',C_0 n^{-3}}(\bar{\boldsymbol{x}}')\Big\rangle - \xi_{\ell-1}(\bar{\boldsymbol{x}}, \bar{\boldsymbol{x}}')\Big| > C\sqrt{d^4 nL}\Big]$$
$$\leq e^{-cd}$$

for any $d \geq K\log n$ and $n \geq K'd^4 L$. Thus, if we set $C_1 = C$ and enforce $d \geq Kd_0\log(nn_0 C_{\mathcal{M}})/\log n$ and $n \geq \max K'd^4 L\log^4 n$, we have by a union bound

$$\mathbb{P}[\mathcal{E}_1 \cup \mathcal{E}_2] \leq n^{-cd}.$$

Let $\mathcal{G} = \mathcal{E}_1^{\mathsf{c}} \cap \mathcal{E}_2^{\mathsf{c}}$. For any $(\boldsymbol{x}, \boldsymbol{x}') \in \mathcal{M} \times \mathcal{M}$, we can find a point $\bar{\boldsymbol{x}} \in N_{n^{-3}n_0^{-1/2}} \cap \mathcal{N}_{n^{-3}n_0^{-1/2}}(\boldsymbol{x})$ and a point $\bar{\boldsymbol{x}}' \in N_{n^{-3}n_0^{-1/2}} \cap \mathcal{N}_{n^{-3}n_0^{-1/2}}(\boldsymbol{x}')$. On $\mathcal{G}$, $\mathsf{SSC}(L, n^{-3}n_0^{-1/2}, Cn^{-3})$ holds at every point in the net $N_{n^{-3}n_0^{-1/2}}$, and there are no more than $d\,Cn^{-3}$-risky features at any point in the net $N_{n^{-3}n_0^{-1/2}}$, and in addition, following (D.52), we have almost surely on $\mathcal{G}$ that all risky features are realized for magnitudes in $(-\Delta, +\Delta)$. This implies that on $\mathcal{G}$, the support sets $\bigsqcup_{\ell \in [L]} I_\ell(\boldsymbol{x})$ at any point $\boldsymbol{x} \in \mathcal{N}_{n^{-3}n_0^{-1/2}}(\bar{\boldsymbol{x}})$ differ by the support sets $\bigsqcup_{\ell \in [L]} I_\ell(\bar{\boldsymbol{x}})$ at the base point in the net by no more than $d$ entries, consisting only of a subset of the risky features at $\bar{\boldsymbol{x}}$; the analogous statement is of course true for $\boldsymbol{x}'$ and $\bar{\boldsymbol{x}}'$. At the same time, notice that on the event $\mathcal{E}_2^{\mathsf{c}}$ we have constructed, we have control of every possible backward feature inner product obtained by modifying the supports at the base points $\bar{\boldsymbol{x}}, \bar{\boldsymbol{x}}'$ at no more than $d$ risky features (each), since, for example, if $(\boldsymbol{\rho}^\ell(\bar{\boldsymbol{x}}))_i$ is risky, then $\mathbb{1}_{(\boldsymbol{\rho}^\ell(\bar{\boldsymbol{x}}))_i > \Delta}$ corresponds to "turning off" the feature, and $\mathbb{1}_{(\boldsymbol{\rho}^\ell(\bar{\boldsymbol{x}}))_i > -\Delta}$ corresponds to "turning on" the feature. Formally, we have established that on $\mathcal{G}$

$$\forall \ell \in [L], \left|\langle \boldsymbol{\beta}^{\ell-1}(\boldsymbol{x}), \boldsymbol{\beta}^{\ell-1}(\boldsymbol{x}')\rangle - \frac{n}{2}\prod_{\ell'=\ell-1}^{L-1}\left(1 - \frac{\varphi^{(\ell')}(\angle(\bar{\boldsymbol{x}}, \bar{\boldsymbol{x}}'))}{\pi}\right)\right| \leq C_1\sqrt{d^4 nL\log^4 n}.$$

We can use differentiability properties for the remaining link: following the proof of Lemma D.10, we have

$$|\angle(\bar{\boldsymbol{x}}, \bar{\boldsymbol{x}}') - \angle(\boldsymbol{x}, \boldsymbol{x}')| \leq \sqrt{2}\|\boldsymbol{x} - \bar{\boldsymbol{x}}\|_2 + \sqrt{2}\|\boldsymbol{x}' - \bar{\boldsymbol{x}}'\|_2 \leq \frac{2\sqrt{2}}{n^3},$$

so we just need a Lipschitz property for the function $q(\nu) = (n/2)\prod_{\ell'=\ell}^{L-1}(1 - \pi^{-1}\varphi^{(\ell')}(\nu))$. For this we appeal to Lemma E.5, which shows that the function $\varphi$ is smooth, increasing and concave; therefore by the chain rule, the functions $\varphi^{(\ell)}$ are increasing and concave, and by the Leibniz rule, $q$ is decreasing and convex. It therefore suffices to calculate $q'(0)$; this is done in Lemma C.18, which gives $q'(0) = -n(L-\ell)/(2\pi)$, and in particular $|q'(0)| \leq cnL$. It follows

$$\left|\frac{n}{2}\prod_{\ell'=\ell}^{L-1}\left(1 - \frac{\varphi^{(\ell')}(\angle(\bar{\boldsymbol{x}}, \bar{\boldsymbol{x}}'))}{\pi}\right) - \frac{n}{2}\prod_{\ell'=\ell}^{L-1}\left(1 - \frac{\varphi^{(\ell')}(\angle(\boldsymbol{x}, \boldsymbol{x}'))}{\pi}\right)\right| \leq \frac{cL}{n^2},$$

so that by the triangle inequality

$$\forall \ell \in [L], \left|\langle \boldsymbol{\beta}^{\ell-1}(\boldsymbol{x}), \boldsymbol{\beta}^{\ell-1}(\boldsymbol{x}')\rangle - \frac{n}{2}\prod_{\ell'=\ell-1}^{L-1}\left(1 - \frac{\varphi^{(\ell')}(\angle(\boldsymbol{x}, \boldsymbol{x}'))}{\pi}\right)\right| \leq 2C_1\sqrt{d^4 nL\log^4 n},$$

where the residual simplification is valid when $n \geq KL$. We conclude that the set

$$\bigcap_{(\boldsymbol{x}, \boldsymbol{x}') \in \mathcal{M} \times \mathcal{M}}\left\{\forall \ell \in [L], \left|\langle \boldsymbol{\beta}^{\ell-1}(\boldsymbol{x}), \boldsymbol{\beta}^{\ell-1}(\boldsymbol{x}')\rangle - \xi_{\ell-1}(\bar{\boldsymbol{x}}, \bar{\boldsymbol{x}}')\right| \leq 2C_1\sqrt{d^4 nL\log^4 n}\right\}$$

contains the event $\mathcal{G}$, which satisfies the claimed properties and completes the proof (after rescaling $d$ by $1/\log n$, which updates the lower bound on $d$). $\qquad\square$

### D.3.4 Small Support Change Residuals

In this section, we prove generalized versions of our pointwise concentration lemmas for backward feature correlations and the matrices defining the propagation coefficients used in our study of $\mathsf{SSC}(L)$.

**Lemma D.14.** *Assume* $n \geq \max\{KL\log n, K'Ld, K''\}$, $d \geq K''' \log L$ *for suitably chosen* $K, K', K'', K'''$ *and integer* $L$, *and choose* $1 \leq \ell' \leq \ell \leq L$. *Define an anchor point* $\boldsymbol{x} \in \mathcal{M}$ *and denote* $I_i(\boldsymbol{x}) = \operatorname{supp}\left(\boldsymbol{\alpha}_0^i(\boldsymbol{x}) > \boldsymbol{0}\right)$ *for* $\ell' \leq i \leq \ell$.

*Choose some* $\delta_s, K_s > 0$ *and let* $\mathcal{J} = \{J_{\ell'}, \ldots, J_\ell\}$ *denote a collection of support sets such that each* $J_i \subset [n]$ *depends on the network parameters only through the pre-activation* $\boldsymbol{\rho}_0^i(\boldsymbol{x})$. *We define events implying that the supports at* $\mathcal{J}$ *are close to those at* $\boldsymbol{x}$:

$$\mathcal{E}_\delta = \bigcap_{\ell' \leq i \leq \ell} \left\{|J_i \ominus I_i(\boldsymbol{x})| \leq \delta_s\right\},$$

$$\mathcal{E}_K = \bigcap_{\ell' \leq i \leq \ell} \left\{\left\|\left(\boldsymbol{P}_{J_i} - \boldsymbol{P}_{I_i(\boldsymbol{x})}\right)\boldsymbol{\rho}^i(\boldsymbol{x})\right\|_2 \leq K_s\right\},$$

$$\mathcal{E}_{\delta K} = \mathcal{E}_\delta \cap \mathcal{E}_K.$$

*Define*

$$\boldsymbol{\Gamma}_{\mathcal{J}}^{\ell:\ell'} = \boldsymbol{P}_{J_\ell}\boldsymbol{W}^\ell \boldsymbol{P}_{J_{\ell-1}} \ldots \boldsymbol{P}_{J_{\ell'}}\boldsymbol{W}^{\ell'},$$

*and fix a unit norm vector* $\boldsymbol{v}_f$. *If* $K_s \leq \frac{1}{2}L^{-3/2}$, $\delta_s \leq \frac{n}{L}$, *then*

$$\mathbb{P}\left[\mathbb{1}_{\mathcal{E}_{\delta K}}\left\|\boldsymbol{\Gamma}_{\mathcal{J}}^{\ell:\ell'}\boldsymbol{v}_f\right\|_2 \leq C\right] \geq 1 - e^{-c\frac{n}{L}},$$

*and*

$$\mathbb{P}\left[\mathbb{1}_{\mathcal{E}_{\delta K}}\left\|\boldsymbol{\Gamma}_{\mathcal{J}}^{\ell:\ell'}\right\| \leq C\sqrt{L}\right] \geq 1 - e^{-c\frac{n}{L}}.$$

*For a vector* $\boldsymbol{g}, g_i \sim_{\text{iid}} \mathcal{N}(0,1)$, *defining* $\boldsymbol{H}^i = \boldsymbol{W}^i\left(\boldsymbol{I} - \boldsymbol{\alpha}^{i-1}(\boldsymbol{x})\boldsymbol{\alpha}^{i-1}(\boldsymbol{x})^*\right)$ *for* $i \in [L]$ *and*

$$\boldsymbol{\Gamma}_{\boldsymbol{H}\mathcal{J}}^{\ell:\ell'} = \boldsymbol{P}_{J_\ell}\boldsymbol{H}^\ell \boldsymbol{P}_{J_{\ell-1}} \ldots \boldsymbol{P}_{J_{\ell'}}\boldsymbol{H}^{\ell'}$$

*we have*

$$\mathbb{P}\left[\mathbb{1}_{\mathcal{E}_{\delta K}}\left\|\left(\boldsymbol{\Gamma}_{\mathcal{J}}^{\ell:\ell'} - \boldsymbol{\Gamma}_{\boldsymbol{H}\mathcal{J}}^{\ell:\ell'}\right)\boldsymbol{g}\right\|_2 > C\sqrt{dL}\right] \leq e^{-cd}$$

*for some numerical constants* $c, C$.

*Proof.* In the following, we will denote by $\boldsymbol{v}_f \in \mathbb{S}^{n-1}$ a fixed unit norm vector and by $\boldsymbol{v}_u \in \mathbb{S}^{n-1}$ a random vector uniformly distributed on $\mathbb{S}^{n-1}$. When there is no need to distinguish between the two we will denote either by $\boldsymbol{v}_p$.

Our strategy in bounding $\mathbb{1}_{\mathcal{E}_{\delta K}}\left\|\boldsymbol{\Gamma}_{\mathcal{J}}^{\ell:\ell'}\right\|$ will be first to bound $\mathbb{1}_{\mathcal{E}_{\delta K}}\left\|\boldsymbol{\Gamma}_{\mathcal{J}}^{\ell:\ell'}\boldsymbol{v}_f\right\|_2$ with sufficiently high probability, and then apply an $\varepsilon$-net argument to uniformize the result (lemma D.20) and get control of the operator norm. In achieving the first goal, we will rely heavily on a decomposition of the weight matrices into terms that are conditionally independent given the pre-activations. We will also utilize martingale concentration to control the terms that result from this decomposition.

Denoting $S^i = \operatorname{span}\left\{\boldsymbol{\alpha}^i(\boldsymbol{x})\right\}$ for $i \in [L]$, we decompose the weight matrices into

$$\boldsymbol{W}^i = \boldsymbol{W}^i \boldsymbol{P}_{S^{i-1}} + \boldsymbol{W}^i \boldsymbol{P}_{S^{i-1\perp}} \doteq \boldsymbol{G}^i + \boldsymbol{H}^i.$$

Note that $\left\{\boldsymbol{H}^1, \ldots, \boldsymbol{H}^L\right\}$ are conditionally independent given $\sigma(\boldsymbol{G}^1, \ldots, \boldsymbol{G}^L)$ (by which we denote the sigma algebra generated by $\boldsymbol{G}^1, \ldots, \boldsymbol{G}^L$). Since the pre-activations obey

$$\boldsymbol{\rho}^i(\boldsymbol{x}) = \boldsymbol{W}^i \boldsymbol{\alpha}^{i-1}(\boldsymbol{x}) = \boldsymbol{G}^i \boldsymbol{\alpha}^{i-1}(\boldsymbol{x})$$

and the features are deterministic functions of the pre-activations, both $\{\boldsymbol{\alpha}^1(\boldsymbol{x}), \ldots, \boldsymbol{\alpha}^L(\boldsymbol{x})\}$ and $\{\boldsymbol{\rho}^1(\boldsymbol{x}), \ldots, \boldsymbol{\rho}^L(\boldsymbol{x})\}$ are measurable with respect to $\sigma(\boldsymbol{G}^1, \ldots, \boldsymbol{G}^L)$.

We define events

$$\mathcal{E}_\rho = \bigcap_{i=\ell'}^{\ell} \left\{ \left\| \boldsymbol{\rho}^i(\boldsymbol{x}) \right\|_2 \leq C \right\}, \quad \mathcal{E}_{\delta K \rho} = \mathcal{E}_{\delta K} \cap \mathcal{E}_\rho \tag{D.78}$$

and aim to control $\mathbb{1}_{\mathcal{E}_{\delta K \rho}} \left\| \boldsymbol{\Gamma}_{\mathcal{J}}^{\ell:\ell'}(\boldsymbol{x}) \right\|$. Since the supports $\mathcal{J}$ depends on the weights only through the pre-activations and are thus also $\sigma(\boldsymbol{G}^1, \ldots, \boldsymbol{G}^L)$-measurable, this truncation does not affect the conditional independence of $\left\{ \boldsymbol{H}^{\ell'}, \ldots, \boldsymbol{H}^\ell \right\}$. It will often be convenient to utilize the rotational invariance of the Gaussian distribution to replace all occurrences of $\boldsymbol{H}^i$ in a given expression by $\tilde{\boldsymbol{W}}^i \boldsymbol{P}_{S^{i-1}\perp}$ where $\tilde{\boldsymbol{W}}^i$ is a fresh copy of $\boldsymbol{W}^i$ independent of all the other variables in the problem, which will not change the distribution of the original expression.

For $\ell' \leq i \leq \ell, \ell' \leq j \leq i+1$ it will also be useful to denote

$$\boldsymbol{\Gamma}_{H\mathcal{J}}^{i:j} = \boldsymbol{P}_{J_i} \boldsymbol{H}^i \boldsymbol{P}_{J_{i-1}} \ldots \boldsymbol{P}_{J_j} \boldsymbol{H}^j, \quad \boldsymbol{\Gamma}_{G\mathcal{J}}^{i:j} = \boldsymbol{P}_{J_i} \boldsymbol{G}^i \boldsymbol{P}_{J_{i-1}} \ldots \boldsymbol{P}_{J_j} \boldsymbol{G}^j$$

where we use the convention $\boldsymbol{\Gamma}_{G\mathcal{J}}^{i:i+1} = \boldsymbol{\Gamma}_{H\mathcal{J}}^{i:i+1} = \boldsymbol{I}$. Decomposing the weight matrices at every layer gives

$$\left\| \boldsymbol{\Gamma}_{\mathcal{J}}^{\ell:\ell'} \boldsymbol{v}_p \right\|_2 = \left\| \boldsymbol{P}_{J_\ell} \left( \boldsymbol{G}^\ell + \boldsymbol{H}^\ell \right) \ldots \boldsymbol{P}_{J_{\ell'}} \left( \boldsymbol{G}^{\ell'} + \boldsymbol{H}^{\ell'} \right) \boldsymbol{v}_p \right\|_2$$

$$\leq \sum_{\left( \boldsymbol{M}^\ell, \ldots, \boldsymbol{M}^{\ell'} \right) \in \left( \boldsymbol{G}^\ell, \boldsymbol{H}^\ell \right) \otimes, \ldots, \otimes \left( \boldsymbol{G}^{\ell'}, \boldsymbol{H}^{\ell'} \right)} \left\| \boldsymbol{P}_{J_\ell} \boldsymbol{M}^\ell \ldots \boldsymbol{P}_{J_{\ell'}} \boldsymbol{M}^{\ell'} \boldsymbol{v}_p \right\|_2. \tag{D.79}$$

We next define

$$\boldsymbol{Q}^i(\boldsymbol{x}) = \boldsymbol{P}_{J_i} - \boldsymbol{P}_{I_i(\boldsymbol{x})}. \tag{D.80}$$

In accounting for all the terms in the decomposition (D.79), there will be two simplifications that we use repeatedly. One is

$$\boldsymbol{H}^{i+1} \boldsymbol{P}_{J_i} \boldsymbol{\rho}^i(\boldsymbol{x}) = \boldsymbol{W}^{i+1} \left( \boldsymbol{I} - \frac{\boldsymbol{\alpha}^i(\boldsymbol{x})\boldsymbol{\alpha}^i(\boldsymbol{x})^*}{\|\boldsymbol{\alpha}^i(\boldsymbol{x})\|_2^2} \right) \left( \boldsymbol{P}_{\boldsymbol{\alpha}^i(\boldsymbol{x})>0} + \boldsymbol{Q}^i(\boldsymbol{x}) \right) \boldsymbol{\rho}^i(\boldsymbol{x}) = \boldsymbol{H}^{i+1} \boldsymbol{Q}^i(\boldsymbol{x})\boldsymbol{\rho}^i(\boldsymbol{x})$$

$$\tag{D.81}$$

where we used $\boldsymbol{P}_{\boldsymbol{\alpha}^i(\boldsymbol{x})>0}\boldsymbol{\rho}^i(\boldsymbol{x}) = \left[ \boldsymbol{\rho}^i(\boldsymbol{x}) \right]_+ = \boldsymbol{\alpha}^i(\boldsymbol{x})$, from which it follows that

$$\boldsymbol{H}^{i+1} \boldsymbol{P}_{J_i} \boldsymbol{G}^i = \boldsymbol{W}^{i+1} \left( \boldsymbol{I} - \frac{\boldsymbol{\alpha}^i(\boldsymbol{x})\boldsymbol{\alpha}^i(\boldsymbol{x})^*}{\|\boldsymbol{\alpha}^i(\boldsymbol{x})\|_2^2} \right) \left( \boldsymbol{P}_{\boldsymbol{\alpha}^i(\boldsymbol{x})>0} + \boldsymbol{Q}^i(\boldsymbol{x}) \right) \frac{\boldsymbol{\rho}^i(\boldsymbol{x})\boldsymbol{\alpha}^{i-1}(\boldsymbol{x})^*}{\|\boldsymbol{\alpha}^{i-1}(\boldsymbol{x})\|_2^2} \tag{D.82}$$

$$= \boldsymbol{H}^{i+1} \boldsymbol{Q}^i(\boldsymbol{x})\boldsymbol{G}^i.$$

We also have

$$\boldsymbol{G}^{i+1} \boldsymbol{P}_{J_i} \boldsymbol{G}^i = \boldsymbol{G}^{i+1} \left( \boldsymbol{P}_{I_i(\boldsymbol{x})} + \boldsymbol{Q}^i(\boldsymbol{x}) \right) \boldsymbol{G}^i$$

$$= \boldsymbol{W}^{i+1} \frac{\boldsymbol{\alpha}^i(\boldsymbol{x})\boldsymbol{\alpha}^i(\boldsymbol{x})^*}{\|\boldsymbol{\alpha}^i(\boldsymbol{x})\|_2^2} \left( \boldsymbol{P}_{I_i(\boldsymbol{x})} + \boldsymbol{Q}^i(\boldsymbol{x}) \right) \boldsymbol{W}^i \frac{\boldsymbol{\alpha}^{i-1}(\boldsymbol{x})\boldsymbol{\alpha}^{i-1}(\boldsymbol{x})^*}{\|\boldsymbol{\alpha}^{i-1}(\boldsymbol{x})\|_2^2}$$

$$= \frac{\|\boldsymbol{\alpha}^i(\boldsymbol{x})\|_2}{\|\boldsymbol{\alpha}^{i-1}(\boldsymbol{x})\|_2} \left( 1 + \frac{\boldsymbol{\alpha}^i(\boldsymbol{x})^*}{\|\boldsymbol{\alpha}^i(\boldsymbol{x})\|_2^2} \boldsymbol{Q}^i(\boldsymbol{x})\boldsymbol{W}^i\boldsymbol{\alpha}^{i-1}(\boldsymbol{x}) \right) \frac{\boldsymbol{W}^{i+1}\boldsymbol{\alpha}^i(\boldsymbol{x})\boldsymbol{\alpha}^{i-1}(\boldsymbol{x})^*}{\|\boldsymbol{\alpha}^i(\boldsymbol{x})\|_2 \|\boldsymbol{\alpha}^{i-1}(\boldsymbol{x})\|_2}$$

$$\doteq s_i \frac{\boldsymbol{W}^{i+1}\boldsymbol{\alpha}^i(\boldsymbol{x})\boldsymbol{\alpha}^{i-1}(\boldsymbol{x})^*}{\|\boldsymbol{\alpha}^i(\boldsymbol{x})\|_2 \|\boldsymbol{\alpha}^{i-1}(\boldsymbol{x})\|_2},$$

and thus

$$\boldsymbol{\Gamma}_{G\mathcal{J}}^{i:j} = \prod_{k=j}^{i-1} s_k \frac{\boldsymbol{P}_{J_i}\boldsymbol{W}^i\boldsymbol{\alpha}^{i-1}(\boldsymbol{x})\boldsymbol{\alpha}^{j-1}(\boldsymbol{x})^*}{\|\boldsymbol{\alpha}^{i-1}(\boldsymbol{x})\|_2 \|\boldsymbol{\alpha}^{j-1}(\boldsymbol{x})\|_2} = \prod_{k=j}^{i-1} s_k \frac{\boldsymbol{P}_{J_i}\boldsymbol{\rho}^i(\boldsymbol{x})\boldsymbol{\alpha}^{j-1}(\boldsymbol{x})^*}{\|\boldsymbol{\alpha}^{i-1}(\boldsymbol{x})\|_2 \|\boldsymbol{\alpha}^{j-1}(\boldsymbol{x})\|_2}. \tag{D.83}$$

We refer to such a product as a G-chain. We proceed to expand (D.79) into terms with different combinations of matrices $\mathbf{\Gamma}_{G\mathcal{J}}^{i:j}$ and $\mathbf{\Gamma}_{H\mathcal{J}}^{i:j}$. There will be $2^{\ell-\ell'}$ terms in total, and we denote the set of terms with $r$ G-chains by $G_{r,p}$ (with the subscript $p \in \{u, f\}$ denoting the choice of vector $\boldsymbol{v}_p$).

We can clearly label each term by the start and end index of each G-chain, which may not be distinct. We denote each such term by $g_{(i_1, i_2, \ldots, i_{2r})}^{rp}$ where

$$\ell' \le i_1 \le i_2 \le i_3 - 2 \le i_4 - 2 < i_5 - 4 \le \cdots \le i_{2m-1} - 2m + 2 \le i_{2m} - 2m + 2 \le \ldots$$
$$\le i_{2r-1} - 2r + 2 \le i_{2r} - 2r + 2 \le \ell - 2r + 2. \tag{D.84}$$

The constraints above ensure that every two G-chains are separated by at least one $\boldsymbol{H}^i$ matrix. To lighten notation, we denote a set of indices obeying the constraints by $(i_1, \ldots, i_{2r}) \in \mathcal{C}_r(\ell, \ell')$. The maximal number of G-chains possible is bounded by $r \le \lceil (\ell - \ell')/2 \rceil$.

Since the $g_{(i_1, i_2, \ldots, i_{2r})}^{rp}$ are non-negative, we have

$$\mathbb{1}_{\mathcal{E}_{\delta K \rho}} \left\| \mathbf{\Gamma}_{\mathcal{J}}^{\ell:\ell'} \boldsymbol{v}_p \right\|_2 \le \mathbb{1}_{\mathcal{E}_{\delta K \rho}} \left\| \mathbf{\Gamma}_{H\mathcal{J}}^{\ell:\ell'} \boldsymbol{v}_p \right\|_2 + \sum_{r=1}^{\lceil (\ell-\ell')/2 \rceil} \sum_{(i_1, \ldots, i_{2r}) \in \mathcal{C}_r(\ell, \ell')} g_{(i_1, i_2, \ldots, i_{2r})}^{r,p}. \tag{D.85}$$

Considering first the form of the terms in $G_{r,p}$, using (D.83) and (D.81) and recalling that $\mathbf{\Gamma}_{H\mathcal{J}}^{m-1:m} = \boldsymbol{I}$, we have

$$
\begin{aligned}
g_{(i_1, i_2, \ldots, i_{2r})}^{r,p} &= \left\| \mathbf{\Gamma}_{H\mathcal{J}}^{\ell:i_{2r}+1} \mathbf{\Gamma}_{G\mathcal{J}}^{i_{2r}:i_{2r-1}} \mathbf{\Gamma}_{H\mathcal{J}}^{i_{2r-1}-1:i_{2r-2}+1} \ldots \mathbf{\Gamma}_{G\mathcal{J}}^{i_4:i_3} \mathbf{\Gamma}_{H\mathcal{J}}^{i_3-1:i_2+1} \mathbf{\Gamma}_{G\mathcal{J}}^{i_2:i_1} \mathbf{\Gamma}_{H\mathcal{J}}^{i_1-1:\ell'} \boldsymbol{v}_p \right\|_2 \\
&= \underbrace{\mathbb{1}_{\mathcal{E}_{\delta K \rho}} \left\| \mathbf{\Gamma}_{H\mathcal{J}}^{\ell:i_{2r}+1} \frac{\boldsymbol{P}_{J^{i_{2r}}} \boldsymbol{\rho}^{i_{2r}}(\boldsymbol{x})}{\|\boldsymbol{\alpha}^{i_{2r}-1}(\boldsymbol{x})\|_2} \right\|_2}_{\doteq \tilde{a}_{i_{2r}}} \\
&\phantom{=} * \prod_{m=1}^{r-1} \underbrace{\mathbb{1}_{\mathcal{E}_{\delta K \rho}} \prod_{k=i_{2m+1}+1}^{i_{2m+2}-1} s_k \frac{\boldsymbol{\alpha}^{i_{2m+1}}(\boldsymbol{x})^* \mathbf{\Gamma}_{H\mathcal{J}}^{i_{2m+1}-1:i_{2m}+1} \boldsymbol{P}_{J^{i_{2m}}} \boldsymbol{\rho}^{i_{2m}}(\boldsymbol{x})}{\|\boldsymbol{\alpha}^{i_{2m+1}}(\boldsymbol{x})\|_2 \|\boldsymbol{\alpha}^{i_{2m}-1}(\boldsymbol{x})\|_2}}_{\doteq \tilde{b}_{i_{2m+2}, i_{2m+1}, i_{2m}}} \\
&\phantom{=} * \underbrace{\mathbb{1}_{\mathcal{E}_{\delta K \rho}} \frac{\prod_{k=i_1+1}^{i_2-1} s_k \boldsymbol{\alpha}^{i_1}(\boldsymbol{x})^* \mathbf{\Gamma}_{H\mathcal{J}}^{i_1-1:\ell'} \boldsymbol{v}_p}{\|\boldsymbol{\alpha}^{i_1}(\boldsymbol{x})\|_2}}_{\doteq \tilde{c}_{i_2, i_1}} \\
&= \tilde{a}_{i_{2r}} \prod_{m=1}^{r-1} \tilde{b}_{i_{2m+2}, i_{2m+1}, i_{2m}} \tilde{c}_{i_2, i_1}^p.
\end{aligned} \tag{D.86}
$$

The magnitudes the factors in this expression are bounded in the following lemma:

**Lemma D.15.** *For $\tilde{a}_k, \tilde{b}_{qij}, \tilde{c}_{ts}^p$ defined in (D.86), $R_u = \sqrt{\frac{d}{n}}, R_f = 1$ and $\ell' \le k < \ell$, $\ell' + 2 \le j + 2 \le i \le q \le \ell$, $\ell' < s \le t \le \ell$*

$$
\begin{aligned}
\tilde{a}_\ell &\le C \ a.s., \\
\mathbb{P}[\tilde{a}_k > K_s] &\le C' e^{-c\frac{n}{L}}, \\
\mathbb{P}\left[\left|\tilde{b}_{qij}\right| > \frac{K_s}{\sqrt{L}}\right] &\le C' e^{-c\frac{n}{L}}, \\
\mathbb{P}[|\tilde{c}_{t\ell'}^p| > CR_p] &\le 2e^{-cd} + e^{-c'n}, \\
\mathbb{P}\left[|\tilde{c}_{ts}^p| > \sqrt{\frac{d}{n}}\right] &\le C' e^{-c\frac{n}{L}} + 2e^{-c'd}
\end{aligned}
$$

*for some constants $c, c', C, C'$ and $d \ge 0$.*

Proof: Deferred to D.3.4.

We will use these results in order to bound $\mathbb{1}_{\mathcal{E}_{\delta K_\rho}} \left\| \boldsymbol{\Gamma}_{\mathcal{J}}^{\ell:\ell'} \boldsymbol{v}_p \right\|_2$ using (D.85). While the sum over most of these terms can be controlled using the triangle inequality and the lemma above, there is a subset which will require special treatment since they are typically larger. These are the terms where the leftmost or rightmost chain is a G-chain (meaning $i_{2r} = \ell$ or $i_1 = \ell'$ respectively) and they will be controlled using martingale concentration. The *or* above is exclusive, since we can bound terms with $i_{2r} = \ell$ *and* $i_1 = \ell'$ using a triangle inequality. We denote these three sets of terms by $\overleftarrow{G}_{r,p}, \overrightarrow{G}_{r,p}, \overleftrightarrow{G}_{r,p}$ respectively, and elements in them by $\overleftarrow{g}^{r,p}, \overrightarrow{g}^{r,p}, \overleftrightarrow{g}^{r,p}$ for clarity when needed. Arranging the remaining terms into sets denoted $\overline{G}_{r,p}$, the sum in (D.85) decomposes into

$$\sum_{r=1}^{\lceil (\ell-\ell')/2 \rceil} \sum_{(i_1,\ldots,i_{2r}) \in \mathcal{C}_r(\ell,\ell')} g_{(i_1,i_2,\ldots,i_{2r})}^r = \sum_{r=1}^{\lceil (\ell-\ell')/2 \rceil} \sum_{Q_{r,p} \in \left\{ \overline{G}_{r,p}, \overleftarrow{G}_{r,p}, \overrightarrow{G}_{r,p}, \overleftrightarrow{G}_{r,p} \right\}} \sum_{g^{r,p} \in Q_{r,p}} g^{r,p}. \tag{D.87}$$

We consider first terms in $\overleftarrow{G}_{r,p}$ (and hence with $i_{2r} = \ell$). We denote such terms by

$$\overleftarrow{g}^{r,p}_{(i_1,i_2,\ldots,i_{2r-1},\ell)} = \tilde{a}_\ell \tilde{b}_{\ell,i_{2r-1},i_{2r-2}} \prod_{m=1}^{r-2} \tilde{b}_{i_{2m+2},i_{2m+1},i_{2m}} \tilde{c}_{i_2,i_1}^p.$$

We show

**Lemma D.16.** *For $p \in \{f, u\}$ and $R_u = \sqrt{\frac{d}{n}}, R_f = 1$*

*i .*

$$\mathbb{P}\left[ \left| \sum_{r=1}^{\lceil (\ell-\ell')/2 \rceil} \sum_{\overleftarrow{g}^{r,p} \in \overleftarrow{G}_{r,p}} \overleftarrow{g}^{r,p} \right| > \sqrt{\frac{dL}{n}} \right] \leq Ce^{-cd} + C'e^{-c'\frac{n}{L}} \tag{D.88}$$

*ii .*

$$\mathbb{P}\left[ \left| \sum_{r=2}^{\lceil (\ell-\ell')/2 \rceil} \sum_{\overrightarrow{g}^{r,p} \in \overrightarrow{G}_{r,p}} \overrightarrow{g}^{r,p} \right| > CR_p \right] \leq Ce^{-cd} + C'e^{-c'\frac{n}{L}} \tag{D.89}$$

*for absolute constants $c, C, C'$, and where $d \geq K \log L$ for some constant $K$.*

Proof: Deferred to D.3.4.

Turning next to bounding the terms in $\overleftrightarrow{G}_{r,p}$, we first define an event

$$\overleftrightarrow{\mathcal{E}} = \{|\tilde{a}_\ell| \leq C\} \cap \bigcap_{\ell'+1 \leq i_1 \leq i_2 - 2 \leq i_3 - 2 \leq \ell-2} \left\{ \left| \tilde{b}_{i_3 i_2 i_1} \right| \leq \frac{K_s}{\sqrt{L}} \right\} \cap \bigcap_{\ell' < i_1 \leq \ell} \{ |\tilde{c}_{i_1 \ell'}^p| \leq CR_p \}$$

and from lemma D.15 and a union bound obtain

$$\mathbb{P}\left[ \overleftrightarrow{\mathcal{E}}^c \right] \leq L^3 C' e^{-c\frac{n}{L}} + L\left( 2e^{-cd} + e^{-c'n} \right) \leq C'' e^{-c'\frac{n}{L}} + 2e^{-c''d}$$

assuming $n \geq KL \log L, d \geq K' \log L$ for some $K, K'$. It follows that

$$
\left| \mathbb{1}_{\overleftrightarrow{\mathcal{E}}} \sum_{\overleftrightarrow{g}_{r,p} \in \overleftrightarrow{G}_{r,p}} \overleftrightarrow{g}_{r,p} \right|
$$

$$
= \left| \mathbb{1}_{\overleftrightarrow{\mathcal{E}}} \sum_{r=1}^{\lceil (\ell-\ell')/2 \rceil} \sum_{(i_2,\ldots,i_{2r-1}) \in \mathcal{C}_{r-1}(\ell',\ell)} \tilde{a}_\ell \tilde{b}_{\ell,i_{2r-1},i_{2r-2}} \prod_{m=1}^{r-2} \tilde{b}_{i_{2m+2},i_{2m+1},i_{2m}} \tilde{c}_{i_2,\ell'} \right|
$$

$$
\leq \sum_{r=1}^{\lceil (\ell-\ell')/2 \rceil} \sum_{(i_2,\ldots,i_{2r-1}) \in \mathcal{C}_{r-1}(\ell',\ell)} \left| \mathbb{1}_{\overleftrightarrow{\mathcal{E}}} \tilde{a}_\ell \tilde{b}_{\ell,i_{2r-1},i_{2r-2}} \prod_{m=1}^{r-2} \tilde{b}_{i_{2m+2},i_{2m+1},i_{2m}} \tilde{c}_{i_2,\ell'} \right|
$$

$$
\leq \sum_{r=1}^{\lceil (\ell-\ell')/2 \rceil} L^{2r-2} \left( \frac{K_s}{\sqrt{L}} \right)^{r-1} R_p
$$

$$
= \sum_{r=1}^{\lceil (\ell-\ell')/2 \rceil} \left( K_s L^{3/2} \right)^{r-1} R_p \leq \frac{1 - \left( K_s L^{3/2} \right)^{L/2}}{1 - K_s L^{3/2}} R_p \leq 2R_p.
$$

where we used $L^{3/2} K_s \leq \frac{1}{2}$. We also bound the number of summands in $\displaystyle\sum_{(i_2,\ldots,i_{2r-1}) \in \mathcal{C}_{r-1}(\ell',\ell)}$ by $L^{2r-2}$ which is tight for small $r$. It follows that

$$
\mathbb{P}\left[ \left| \sum_{\overleftrightarrow{g}_{r,p} \in \overleftrightarrow{G}_{r,p}} \overleftrightarrow{g}_{r,p} \right| > 2R_p \right] \leq \mathbb{P}\left[ \left| \mathbb{1}_{\overleftrightarrow{\mathcal{E}}} \sum_{\overleftrightarrow{g}_{r,p} \in \overleftrightarrow{G}_{r,p}} \overleftrightarrow{g}_{r,p} \right| > 2R_p \right]
$$

$$
+ \mathbb{P}\left[ \left| (1 - \mathbb{1}_{\overleftrightarrow{\mathcal{E}}}) \sum_{\overleftrightarrow{g}_{r,p} \in \overleftrightarrow{G}_{r,p}} \overleftrightarrow{g}_{r,p} \right| > 0 \right]
$$

$$
= \mathbb{P}\left[ \overleftrightarrow{\mathcal{E}}^c \right] \leq C e^{-c \frac{n}{L}} + 2e^{-c'd} \tag{D.90}
$$

for appropriate constants. It remains to bound the terms in $\overline{G}_{r,p}$ by a similar argument. Defining

$$
\overline{\mathcal{E}} = \bigcap_{\ell' \leq i_1 < \ell} \{|\tilde{a}_{i_1}| \leq K_s\} \cap \bigcap_{\ell'+2 \leq i_1+2 \leq i_2 \leq i_3 \leq \ell} \left\{ \left| \tilde{b}_{i_3 i_2 i_1} \right| \leq \frac{K_s}{\sqrt{L}} \right\}
$$

$$
\cap \bigcap_{\ell' < i_1 \leq i_2 \leq \ell} \left\{ \left| \tilde{c}_{i_2 i_1}^p \right| \leq \sqrt{\frac{d}{n}} \right\}
$$

truncating on this event gives

$$
\left| \mathbb{1}_{\overline{\mathcal{E}}} \sum_{\overline{g}_{r,p} \in \overline{G}_{r,p}} \overline{g}_{r,p} \right| \leq \mathbb{1}_{\overline{\mathcal{E}}} \sum_{\overline{g}_{r,p} \in \overline{G}_{r,p}} |\overline{g}_{r,p}| \leq \left( L^{3/2} K_s \right)^r \sqrt{\frac{dL}{n}} \leq 2\sqrt{\frac{dL}{n}}
$$

and bounding the probability of this even from below using lemma D.15 and a union bound gives

$$
\mathbb{P}\left[ \left| \sum_{\overline{g}_{r,p} \in \overline{G}_{r,p}} \overline{g}_{r,p} \right| > 2\sqrt{\frac{dL}{n}} \right] \leq \mathbb{P}\left[ \left| \mathbb{1}_{\overline{\mathcal{E}}} \sum_{\overline{g}_{r,p} \in \overline{G}_{r,p}} \overline{g}_{r,p} \right| > 2\sqrt{\frac{dL}{n}} \right] + \mathbb{P}\left[ \overline{\mathcal{E}}^c \right]
$$

$$
\leq C e^{-c \frac{n}{L}} + 2e^{-c'd}.
$$

Combining the bound above with (D.90) and the results of lemma D.16 and worsening constants, the sum of all terms containing matrices $\boldsymbol{G}^i$ is bounded by

$$
\mathbb{P}\left[\left|\sum_{g_{r,p} \in G_{r,p}} g_{r,p}\right| > C\left(\sqrt{\frac{dL}{n}} + R_p\right)\right] \leq
\begin{aligned}
&\sum_{Q_{r,p} \in \left\{\overline{G}_{r,p}, \overleftarrow{G}_{r,p}\right\}} \mathbb{P}\left[\left|\sum_{g^{r,p} \in Q_{r,p}} g^{r,p}\right| > C\sqrt{\frac{dL}{n}}\right] \\
&+ \sum_{Q_{r,p} \in \left\{\overrightarrow{G}_{r,p}, \overleftrightarrow{G}_{r,p}\right\}} \mathbb{P}\left[\left|\sum_{g^{r,p} \in Q_{r,p}} g^{r,p}\right| > CR_p\right]
\end{aligned}
$$

$$
\leq C'e^{-c\frac{n}{L}} + C''e^{-c'd}.
$$

$$(D.91)$$

Bounding the first term in (D.85) using lemma D.18, setting $p = f$ above and choosing $d = \frac{n}{L}$ gives

$$
\mathbb{P}\left[\mathbb{1}_{\mathcal{E}_{\delta K \rho}} \left\|\boldsymbol{\Gamma}_{\mathcal{J}}^{\ell:\ell'} \boldsymbol{v}_f\right\|_2 > C\right] \leq C'e^{-c'\frac{n}{L}}.
$$

We then apply lemma D.20 to obtain

$$
\mathbb{P}\left[\mathbb{1}_{\mathcal{E}_{\delta K \rho}} \left\|\boldsymbol{\Gamma}_{\mathcal{J}}^{\ell:\ell'}\right\| > C\sqrt{L}\right] \leq C'e^{-c'\frac{n}{L}}.
$$

Recalling (D.78), to obtain our final bound on the operator norm it remains to control the probability of $\mathcal{E}_\rho$. We consider some $\ell' \leq i \leq \ell$ and assume $\boldsymbol{\alpha}^{i-1}(\boldsymbol{x}) \neq 0$ (otherwise $\left\|\boldsymbol{\rho}^i(\boldsymbol{x})\right\|_2 \leq C$ with probability 1). Defining an orthogonal matrix $\boldsymbol{R}$ such that $\boldsymbol{R}\boldsymbol{\alpha}^{i-1}(\boldsymbol{x}) = \left\|\boldsymbol{\alpha}^{i-1}(\boldsymbol{x})\right\|_2 \widehat{\boldsymbol{e}}_1$, rotational invariance of the Gaussian distribution gives

$$
\left\|\boldsymbol{\rho}^i(\boldsymbol{x})\right\|_2^2 = \boldsymbol{\alpha}^{i-1}(\boldsymbol{x})^* \boldsymbol{W}^{i*} \boldsymbol{W}^i \boldsymbol{\alpha}^{i-1}(\boldsymbol{x}) \overset{d}{=} \left\|\boldsymbol{\alpha}^{i-1}(\boldsymbol{x})\right\|_2^2 \left\|\boldsymbol{W}_{(:,1)}^i\right\|_2^2,
$$

$$
\underset{\boldsymbol{W}^i}{\mathbb{E}} \left\|\boldsymbol{\rho}^i(\boldsymbol{x})\right\|_2^2 = 2\left\|\boldsymbol{\alpha}^{i-1}(\boldsymbol{x})\right\|_2^2.
$$

Since $\left\|\boldsymbol{W}_{(:,1)}^i\right\|_2^2$ is a sum of independent sub-exponential random variables with sub-exponential norm bounded by $\frac{C'}{n}$, Bernstein's inequality (lemma G.2) and D.2 give

$$
\mathbb{P}\left[\left\|\boldsymbol{\rho}^i(\boldsymbol{x})\right\|_2^2 > C\right] \leq \mathbb{P}\left[\left\|\boldsymbol{\rho}^i(\boldsymbol{x})\right\|_2^2 - \left\|\boldsymbol{\alpha}^{i-1}(\boldsymbol{x})\right\|_2^2 > \frac{C}{2} \middle| \left\|\boldsymbol{\alpha}^{i-1}(\boldsymbol{x})\right\|_2^2 \leq \frac{C}{2}\right]
$$

$$
+ \mathbb{P}\left[\left\|\boldsymbol{\alpha}^{i-1}(\boldsymbol{x})\right\|_2^2 > \frac{C}{2}\right]
$$

$$
\leq 2e^{-cn} + C'e^{-c\frac{n}{L}} \leq C''e^{-c'\frac{n}{L}}
$$

for appropriate constants. Taking a union bound over $i$ gives

$$
\mathbb{P}\left[\mathcal{E}_\rho\right] \geq 1 - C''Le^{-c'\frac{n}{L}} \geq 1 - C''e^{-c''\frac{n}{L}}
$$

for a suitable chosen constant $c''$, where we used $n \geq KL\log L$ for some $K$.

We then have

$$
\mathbb{P}\left[\mathbb{1}_{\mathcal{E}_{\delta K}} \left\|\boldsymbol{\Gamma}_{\mathcal{J}}^{\ell:\ell'} \boldsymbol{v}_f\right\|_2 > C\right] \leq \mathbb{P}\left[\mathbb{1}_{\mathcal{E}_{\delta K} \cap \mathcal{E}_\rho} \left\|\boldsymbol{\Gamma}_{\mathcal{J}}^{\ell:\ell'} \boldsymbol{v}_f\right\|_2 > C\right] + \mathbb{P}\left[\mathbb{1}_{\mathcal{E}_{\delta K} \cap \mathcal{E}_\rho^c} \left\|\boldsymbol{\Gamma}_{\mathcal{J}}^{\ell:\ell'} \boldsymbol{v}_f\right\|_2 > 0\right]
$$

$$
\leq \mathbb{P}\left[\mathbb{1}_{\mathcal{E}_{\delta K} \cap \mathcal{E}_\rho} \left\|\boldsymbol{\Gamma}_{\mathcal{J}}^{\ell:\ell'} \boldsymbol{v}_f\right\|_2 > C\right] + \mathbb{P}\left[\mathcal{E}_\rho^c\right] \leq Ce^{-c\frac{n}{L}} + C'e^{-c'\frac{n}{L}} \leq C''e^{-c''\frac{n}{L}}
$$

for appropriate constants, and similarly

$$
\mathbb{P}\left[\mathbb{1}_{\mathcal{E}_{\delta K}} \left\|\boldsymbol{\Gamma}_{\mathcal{J}}^{\ell:\ell'}\right\| > C\sqrt{L}\right] \leq C''e^{-c''\frac{n}{L}}.
$$

This concludes the proof of the first two statements. For the final result, we consider a vector $\boldsymbol{g}$ with $g_i \sim_{\text{iid}} \mathcal{N}(0,1)$. Bernstein's inequality gives $\mathbb{P}\left[\|\boldsymbol{g}\|_2^2 > 2n\right] \leq e^{-cn}$ and since

$$
\boldsymbol{g} \overset{d}{=} \boldsymbol{v}_u \|\boldsymbol{g}\|_2
$$

where $\boldsymbol{v}_u$ is uniformly distributed on $\mathbb{S}^{n-1}$, we can use D.91 setting $p = u$ to obtain

$$\mathbb{P}\left[\left\|\left(\boldsymbol{\Gamma}_{\mathcal{J}}^{L:\ell}(\boldsymbol{x}) - \boldsymbol{\Gamma}_{H\mathcal{J}}^{L:\ell}(\boldsymbol{x})\right)\boldsymbol{g}\right\|_2 > C\sqrt{dL}\right] \leq \mathbb{P}\left[\left\|\left(\boldsymbol{\Gamma}_{\mathcal{J}}^{L:\ell}(\boldsymbol{x}) - \boldsymbol{\Gamma}_{H\mathcal{J}}^{L:\ell}(\boldsymbol{x})\right)\boldsymbol{v}_u\right\|_2 > \frac{C}{2}\sqrt{\frac{dL}{n}}\right]$$

$$+ \mathbb{P}\left[\|\boldsymbol{g}\|_2 > 2\sqrt{n}\right]$$

$$\leq e^{-cn} + C'e^{-c'd} + C''e^{-c''\frac{n}{L}} \leq C'''e^{-c'''d}.$$

for appropriate constants, where we assumed $n > KLd$ for some $K$. $\qquad\square$

**Corollary D.17.** *Defining*

$$\boldsymbol{\Gamma}^{\ell:\ell'}(\boldsymbol{x}) = \boldsymbol{P}_{I_\ell}\boldsymbol{W}^\ell\boldsymbol{P}_{I_{\ell-1}}\ldots\boldsymbol{P}_{I_{\ell'}}\boldsymbol{W}^{\ell'},$$

*under the same assumptions on $n, L$ in D.14 we have*

$$\mathbb{P}\left[\left\|\boldsymbol{\Gamma}^{\ell:\ell'}(\boldsymbol{x})\boldsymbol{v}\right\|_2 \leq C\right] \geq 1 - e^{-c\frac{n}{L}}$$

$$\mathbb{P}\left[\left\|\boldsymbol{\Gamma}^{\ell:\ell'}(\boldsymbol{x})\right\| \leq C\sqrt{L}\right] \geq 1 - e^{-c\frac{n}{L}}$$

*for some numerical constants $c, C$.*

**Lemma D.18.** *Fix a collection of supports $\mathcal{J} = \{J_{\ell'}\ldots J_\ell\}$ for $1 \leq \ell' \leq \ell \leq L$ that satisfy the assumptions of lemma D.14 and denote by $\boldsymbol{v}_p$ a unit norm vector. Define an event*

$$\mathcal{E}_H^{\ell\ell'} = \left\{\mathbb{1}_{\mathcal{E}_\delta}\left\|\boldsymbol{\Gamma}_{H\mathcal{J}}^{\ell:\ell'}\boldsymbol{v}_p\right\|_2 \leq C\right\} \cap \left\{\mathbb{1}_{\mathcal{E}_\delta}\left\|\boldsymbol{\Gamma}_{H\mathcal{J}}^{\ell:\ell'}\right\| \leq C\sqrt{L}\right\} \cap \left\{\mathbb{1}_{\mathcal{E}_\delta}\left\|\boldsymbol{\Gamma}_{H\mathcal{J}}^{\ell:\ell'}\right\|_F^2 \leq Cn\right\}.$$

*If $n \geq KL\log n$ for some constant $K$ then*

$$\mathbb{P}\left[\mathcal{E}_H^{\ell\ell'}\right] \geq 1 - C'e^{-c\frac{n}{L}}$$

*where $c, C, C'$ are absolute constants.*

*Proof.* In the following, we denote by $\tilde{\boldsymbol{W}}^i$ an independent copy of $\boldsymbol{W}^i$, and by $\boldsymbol{W}_{(:,j)}^i$ the $j$-th column of this matrix. Note that due to rotational invariance of the Gaussian distribution we can replace every occurrence of $\boldsymbol{H}^i$ in an expression by $\tilde{\boldsymbol{W}}^i\boldsymbol{P}_{S^{i-1}\perp}$ without altering the distribution of the expression, which we will do presently. We can repeatedly use this rotational invariance to give

$$\boldsymbol{v}_p^*\boldsymbol{\Gamma}_{H\mathcal{J}}^{\ell:\ell'*}\boldsymbol{\Gamma}_{H\mathcal{J}}^{\ell:\ell'}\boldsymbol{v}_p = \boldsymbol{v}_p^*\boldsymbol{H}^{\ell'*}\boldsymbol{P}_{J_{\ell'}}\boldsymbol{\Gamma}_{H\mathcal{J}}^{\ell:\ell'+1*}\boldsymbol{\Gamma}_{H\mathcal{J}}^{\ell:\ell'+1}\boldsymbol{P}_{J_{\ell'}}\boldsymbol{H}^{\ell'}\boldsymbol{v}_p$$

$$\stackrel{d}{=} \boldsymbol{v}_p^*\boldsymbol{P}_{S^{\ell'-1}\perp}\tilde{\boldsymbol{W}}^{\ell'*}\boldsymbol{P}_{J_{\ell'}}\boldsymbol{\Gamma}_{H\mathcal{J}}^{\ell:\ell'+1*}\boldsymbol{\Gamma}_{H\mathcal{J}}^{\ell:\ell'+1}\boldsymbol{P}_{J_{\ell'}}\tilde{\boldsymbol{W}}^{\ell'}\boldsymbol{P}_{S^{\ell'-1}\perp}\boldsymbol{v}_p$$

and rotational invariance of the Gaussian distribution gives

$$\boldsymbol{v}_p^*\boldsymbol{\Gamma}_{H\mathcal{J}}^{\ell:\ell'*}\boldsymbol{\Gamma}_{H\mathcal{J}}^{\ell:\ell'}\boldsymbol{v}_p \stackrel{d}{=} \|\boldsymbol{P}_{S^{\ell'\perp}}\boldsymbol{v}_p\|_2^2\tilde{\boldsymbol{W}}_{(:,1)}^{\ell'*}\boldsymbol{P}_{J_{\ell'}}\boldsymbol{\Gamma}_{H\mathcal{J}}^{\ell:\ell'+1*}\boldsymbol{\Gamma}_{H\mathcal{J}}^{\ell:\ell'+1}\boldsymbol{P}_{J_{\ell'}}\tilde{\boldsymbol{W}}_{(:,1)}^{\ell'}$$

$$\stackrel{d}{=} \|\boldsymbol{P}_{S^{\ell'\perp}}\boldsymbol{v}_p\|_2^2\tilde{\boldsymbol{W}}_{(:,1)}^{\ell'*}\boldsymbol{P}_{J_{\ell'}}\boldsymbol{P}_{S^{\ell'+1\perp}}\tilde{\boldsymbol{W}}^{\ell'+1*}\boldsymbol{\Gamma}_{H\mathcal{J}}^{\ell:\ell'+2*}\boldsymbol{\Gamma}_{H\mathcal{J}}^{\ell:\ell'+2}\boldsymbol{P}_{J_{\ell'+1}}\tilde{\boldsymbol{W}}^{\ell'+1}\boldsymbol{P}_{S^{\ell'+1\perp}}\boldsymbol{P}_{J_{\ell'}}\tilde{\boldsymbol{W}}_{(:,1)}^{\ell'}$$

$$\stackrel{d}{=} \|\boldsymbol{P}_{S^{\ell'\perp}}\boldsymbol{v}_p\|_2^2\left\|\boldsymbol{P}_{S^{\ell'+1\perp}}\boldsymbol{P}_{J_{\ell'}}\tilde{\boldsymbol{W}}_{(:,1)}^{\ell'}\right\|_2^2\tilde{\boldsymbol{W}}_{(:,1)}^{\ell'+1*}\boldsymbol{P}_{J_{\ell'+1}}\boldsymbol{\Gamma}_{H\mathcal{J}}^{\ell:\ell'+2*}\boldsymbol{\Gamma}_{H\mathcal{J}}^{\ell:\ell'+2}\boldsymbol{P}_{J_{\ell'+1}}\tilde{\boldsymbol{W}}_{(:,1)}^{\ell'+1}$$

and continuing in a similar fashion we obtain

$$\boldsymbol{v}_p^*\boldsymbol{\Gamma}_{H\mathcal{J}}^{\ell:\ell'*}\boldsymbol{\Gamma}_{H\mathcal{J}}^{\ell:\ell'}\boldsymbol{v}_p \stackrel{d}{=} \|\boldsymbol{P}_{S^{\ell'\perp}}\boldsymbol{v}_p\|_2^2\prod_{i=\ell'}^{\ell-1}\left\|\boldsymbol{P}_{S^{i+1\perp}}\boldsymbol{P}_{J_i}\tilde{\boldsymbol{W}}_{(:,1)}^i\right\|_2^2\left\|\boldsymbol{P}_{J_\ell}\tilde{\boldsymbol{W}}_{(:,1)}^\ell\right\|_2^2$$

$$\leq \prod_{i=\ell'}^{\ell}\left\|\boldsymbol{P}_{J_i}\tilde{\boldsymbol{W}}_{(:,1)}^i\right\|_2^2 \quad a.s.$$

where in the last inequality we used the fact that multiplication by $\boldsymbol{P}_{S^{i\perp}}$ cannot increase the norm of a vector. Denoting by $\{\chi_i\}$ a collection of independent standard chi-squared distributed random variables where $\chi_i$ has $|J_i|$ degrees of freedom, we have

$$\prod_{i=\ell'}^{\ell} \left\| \boldsymbol{P}_{J_i} \tilde{\boldsymbol{W}}_{(:,1)}^i \right\|_2^2 \stackrel{d}{=} \prod_{i=\ell'}^{\ell} \frac{2}{n} \chi_i.$$

Define

$$\mathcal{E}_I = \left\{ \min_{\ell' \leq i \leq \ell} |I_i(\boldsymbol{x})| \geq \frac{n}{4} \right\} \cap \left\{ \prod_{i=\ell'}^{\ell} \frac{2|I_i(\boldsymbol{x})|}{n} \leq 2 \right\}.$$

Denoting $\delta^i = |J_i \ominus I_i(\boldsymbol{x})|$, on $\mathcal{E}_I$ we have

$$\min_{\ell' \leq i \leq \ell} |J_i| \geq \frac{n}{4} - \frac{n}{L} \geq \frac{n}{8}. \tag{D.92}$$

$$\prod_{i=\ell'}^{\ell} \frac{2|J_i|}{n} \leq \prod_{i=\ell'}^{\ell} \frac{2\left(|I_i| + \delta^i\right)}{n} \leq \prod_{i=\ell'}^{\ell} \frac{2\left(|I_i| + \frac{n}{L}\right)}{n} = \prod_{i=\ell'}^{\ell} \frac{2|I_i|}{n} \left(1 + \frac{n}{L|I_i|}\right)$$

$$\leq \prod_{i=\ell'}^{\ell} \frac{2|I_i|}{n} \left(1 + \frac{4}{L}\right) \leq e^4 \prod_{i=\ell'}^{\ell} \frac{2|I_i|}{n} \leq 2e^4.$$

where we used the assumption $\delta^i \leq \frac{n}{L}$ and assumed $n \geq 8L$. It follows that

$$\mathbb{P}\left[ \left| \prod_{i=\ell'}^{\ell} \frac{2}{n}\chi_i - \prod_{i=\ell'}^{\ell} \frac{2|J_i|}{n} \right| > 1 \,\middle|\, \mathcal{E}_\delta \cap \mathcal{E}_I \right] = \mathbb{P}\left[ \left| \prod_{i=\ell'}^{\ell} \frac{\chi_i}{|J_i|} - 1 \right| > \prod_{i=\ell'}^{\ell} \frac{n}{2|J_i|} \,\middle|\, \mathcal{E}_\delta \cap \mathcal{E}_I \right]$$

$$\leq \mathbb{P}\left[ \left| \prod_{i=\ell'}^{\ell} \frac{\chi_i}{|J_i|} - 1 \right| > \frac{1}{2e^4} \,\middle|\, \mathcal{E}_\delta \cap \mathcal{E}_I \right].$$

An application of lemma D.26 and (D.92) then gives

$$\mathbb{P}\left[ \left| \prod_{i=\ell'}^{\ell} \frac{\chi_i}{|J_i|} - 1 \right| > \frac{1}{2e^4} \,\middle|\, \mathcal{E}_\delta \cap \mathcal{E}_I \right] \leq CLe^{-c\frac{n}{L}} \leq Ce^{-c'\frac{n}{L}} \tag{D.93}$$

for appropriate constants, assuming $n \geq KL \log L$ for some $K$. Using D.30 to bound $\mathbb{P}\left[\mathcal{E}_I^c\right]$ we thus have

$$\mathbb{P}\left[ \mathbb{1}_{\mathcal{E}_\delta} \boldsymbol{v}_p^* \boldsymbol{\Gamma}_{\boldsymbol{HJ}}^{\ell:\ell'*} \boldsymbol{\Gamma}_{\boldsymbol{HJ}}^{\ell:\ell'} \boldsymbol{v}_p > 1 + 2e^4 \right] \leq \mathbb{P}\left[ \mathbb{1}_{\mathcal{E}_\delta} \prod_{i=\ell'}^{\ell} \frac{2}{n}\chi_i > 1 + \prod_{i=\ell'}^{\ell} \frac{2|J_i|}{n} \right]$$

$$\leq \mathbb{P}\left[ \prod_{i=\ell'}^{\ell} \frac{2}{n}\chi_i > 1 + \prod_{i=\ell'}^{\ell} \frac{2|J_i|}{n} \,\middle|\, \mathcal{E}_\delta \right]$$

$$\leq \mathbb{P}\left[ \prod_{i=\ell'}^{\ell} \frac{2}{n}\chi_i > 1 + \prod_{i=\ell'}^{\ell} \frac{2|J_i|}{n} \,\middle|\, \mathcal{E}_\delta \cap \mathcal{E}_I \right] + \mathbb{P}\left[\mathcal{E}_I^c\right]$$

$$\leq Ce^{-c\frac{n}{L}} + C'e^{-c'\frac{n}{L}} \leq C''e^{-c''\frac{n}{L}}$$

for some constants. Having shown

$$\mathbb{P}\left[ \mathbb{1}_{\mathcal{E}_\delta} \boldsymbol{v}_p^* \boldsymbol{\Gamma}_{\boldsymbol{HJ}}^{\ell:\ell'*} \boldsymbol{\Gamma}_{\boldsymbol{HJ}}^{\ell:\ell'} \boldsymbol{v}_p > C \right] \leq C'e^{-c\frac{n}{L}}$$

for some fixed $\boldsymbol{v}_p = \boldsymbol{v}_f$ we can now apply lemma D.20 to obtain

$$\mathbb{P}\left[ \mathbb{1}_{\mathcal{E}_\delta} \left\| \boldsymbol{\Gamma}_{\boldsymbol{HJ}}^{\ell:\ell'} \right\| > C\sqrt{L} \right] \leq C''Le^{-c'\frac{n}{L}} \leq C''e^{-c''\frac{n}{L}}.$$

where we used $n \geq KL \log L$ for some $K$. Choosing $\boldsymbol{v}_p = \widehat{\boldsymbol{e}}_i$ for $i \in [n]$ and taking a union bound, one obtains

$$\mathbb{P}\left[\mathbb{1}_{\mathcal{E}_\delta}\left\|\boldsymbol{\Gamma}_{\boldsymbol{HJ}}^{\ell:\ell'}\right\|_F^2 > Cn\right] = \mathbb{P}\left[\mathbb{1}_{\mathcal{E}_\delta}\mathrm{tr}\left[\boldsymbol{\Gamma}_{\boldsymbol{HJ}}^{\ell:\ell'*}\boldsymbol{\Gamma}_{\boldsymbol{HJ}}^{\ell:\ell'}\right] > Cn\right] = \mathbb{P}\left[\sum_{i=1}^n \mathbb{1}_{\mathcal{E}_\delta}\widehat{\boldsymbol{e}}_i^*\boldsymbol{\Gamma}_{\boldsymbol{HJ}}^{\ell:\ell'*}\boldsymbol{\Gamma}_{\boldsymbol{HJ}}^{\ell:\ell'}\widehat{\boldsymbol{e}}_i > Cn\right]$$

$$\leq \sum_{i=1}^n \mathbb{P}\left[\mathbb{1}_{\mathcal{E}_\delta}\widehat{\boldsymbol{e}}_i^*\boldsymbol{\Gamma}_{\boldsymbol{HJ}}^{\ell:\ell'*}\boldsymbol{\Gamma}_{\boldsymbol{HJ}}^{\ell:\ell'}\widehat{\boldsymbol{e}}_i > C\right] \leq nC''e^{-c''\frac{n}{L}} \leq C'e^{-c\frac{n}{L}}$$

for some constants, where we used $n \geq KL \log n$ for an appropriate constant $K$. A final union bound over the last three events gives the desired result. $\qquad\square$

*Proof of lemma D.15.* We first consider the terms $\tilde{a}_k$. For $k = \ell$, the definition of $\mathcal{E}_{\delta K\rho}$ in (D.78) gives

$$\tilde{a}_\ell = \mathbb{1}_{\mathcal{E}_{\delta K\rho}}\frac{\left\|\boldsymbol{\Gamma}_{\boldsymbol{HJ}}^{\ell:\ell+1}\boldsymbol{P}_{J_\ell}\boldsymbol{\rho}^\ell(\boldsymbol{x})\right\|_2}{\left\|\boldsymbol{\alpha}^{\ell-1}(\boldsymbol{x})\right\|_2} = \mathbb{1}_{\mathcal{E}_{\delta K\rho}}\frac{\left\|\boldsymbol{P}_{J_\ell}\boldsymbol{\rho}^\ell(\boldsymbol{x})\right\|_2}{\left\|\boldsymbol{\alpha}^{\ell-1}(\boldsymbol{x})\right\|_2} \leq C \; a.s. \tag{D.94}$$

In order to handle the case $\ell' \leq k < \ell$, we use (D.81) and obtain that for any $2 \leq j \leq i \leq L$,

$$\boldsymbol{\Gamma}_{\boldsymbol{HJ}}^{i:j}\boldsymbol{P}_{J_{j-1}}\frac{\boldsymbol{\rho}^{j-1}(\boldsymbol{x})}{\left\|\boldsymbol{\alpha}^{j-2}(\boldsymbol{x})\right\|_2} = \boldsymbol{\Gamma}_{\boldsymbol{HJ}}^{i:j+1}\boldsymbol{P}_{J_j}\boldsymbol{H}^j\boldsymbol{P}_{J_{j-1}}\frac{\boldsymbol{\rho}^{j-1}(\boldsymbol{x})}{\left\|\boldsymbol{\alpha}^{j-2}(\boldsymbol{x})\right\|_2}$$

$$= \boldsymbol{\Gamma}_{\boldsymbol{HJ}}^{i:j+1}\boldsymbol{P}_{J_j}\boldsymbol{H}^j\boldsymbol{Q}^{j-1}(\boldsymbol{x})\frac{\boldsymbol{\rho}^{j-1}(\boldsymbol{x})}{\left\|\boldsymbol{\alpha}^{j-2}(\boldsymbol{x})\right\|_2}$$

$$\overset{d}{=} \boldsymbol{\Gamma}_{\boldsymbol{HJ}}^{i:j+1}\boldsymbol{P}_{J_j}\tilde{\boldsymbol{W}}^j\boldsymbol{P}_{S^{j-1\perp}}\boldsymbol{Q}^{j-1}(\boldsymbol{x})\boldsymbol{\rho}^{j-1}(\boldsymbol{x})$$

$$\overset{d}{=} \boldsymbol{\Gamma}_{\boldsymbol{HJ}}^{i:j+1}\boldsymbol{P}_{J_j}\tilde{\boldsymbol{W}}_{(:,1)}^j\frac{\left\|\boldsymbol{P}_{S^{j-1\perp}}\boldsymbol{Q}^{j-1}(\boldsymbol{x})\boldsymbol{\rho}^{j-1}(\boldsymbol{x})\right\|_2}{\left\|\boldsymbol{\alpha}^{j-2}(\boldsymbol{x})\right\|_2}$$

where $\tilde{\boldsymbol{W}}^j$ is an independent copy of $\boldsymbol{W}^j$, and we denote by $\tilde{\boldsymbol{W}}_{(:,1)}^j$ the first column of $\tilde{\boldsymbol{W}}^j$, and we used the rotational invariance of the Gaussian distribution. Truncating on the event $\mathcal{E}_H^{i,j+1}$, which does not affect the distribution of $\tilde{\boldsymbol{W}}_{(:,1)}^j$, we have

$$\underset{\tilde{\boldsymbol{W}}_{(:,1)}^j}{\mathbb{E}}\left\|\mathbb{1}_{\mathcal{E}_H^{i,j+1}}\boldsymbol{\Gamma}_{\boldsymbol{HJ}}^{i:j+1}\boldsymbol{P}_{J_j}\tilde{\boldsymbol{W}}_{(:,1)}^j\right\|_2^2 = \frac{2}{n}\mathbb{1}_{\mathcal{E}_H^{i,j+1}}\mathrm{tr}\left[\boldsymbol{P}_{J_j}\boldsymbol{\Gamma}_{\boldsymbol{HJ}}^{i:j+1*}\boldsymbol{\Gamma}_{\boldsymbol{HJ}}^{i:j+1}\right]$$

$$\leq \frac{2}{n}\mathbb{1}_{\mathcal{E}_H^{i,j+1}}\mathrm{tr}\left[\boldsymbol{\Gamma}_{\boldsymbol{HJ}}^{i:j+1*}\boldsymbol{\Gamma}_{\boldsymbol{HJ}}^{i:j+1}\right]$$

$$= \frac{2}{n}\mathbb{1}_{\mathcal{E}_H^{i,j+1}}\left\|\boldsymbol{\Gamma}_{\boldsymbol{HJ}}^{i:j+1}\right\|_F^2 \leq C'$$

almost surely for some constant $C'$, and the Hanson-Wright inequality (lemma G.4) gives

$$\mathbb{P}\left[\left\|\mathbb{1}_{\mathcal{E}_H^{i,j+1}}\boldsymbol{\Gamma}_{\boldsymbol{HJ}}^{i:j+1}\boldsymbol{P}_{J_j}\tilde{\boldsymbol{W}}_{(:,1)}^j\right\|_2^2 > 1 + C'\right]$$

$$\leq 2\exp\left(-c\min\left\{\frac{n^2}{4\left\|\mathbb{1}_{\mathcal{E}_H^{i,j+1}}\boldsymbol{\Gamma}_{\boldsymbol{HJ}}^{i:j+1*}\boldsymbol{\Gamma}_{\boldsymbol{HJ}}^{i:j+1}\right\|_F^2}, \frac{n}{2\left\|\mathbb{1}_{\mathcal{E}_H^{i,j+1}}\boldsymbol{\Gamma}_{\boldsymbol{HJ}}^{i:j+1*}\boldsymbol{\Gamma}_{\boldsymbol{HJ}}^{i:j+1}\right\|}\right\}\right)$$

$$\leq 2\exp\left(-c'\frac{n}{(i-j+1)}\right)$$

for some constant $c'$, where we used $\left\|\mathbb{1}_{\mathcal{E}_H^{i,j+1}}\boldsymbol{\Gamma}_{\boldsymbol{HJ}}^{i:j+1*}\boldsymbol{\Gamma}_{\boldsymbol{HJ}}^{i:j+1}\right\|_F^2 \leq \left\|\mathbb{1}_{\mathcal{E}_H^{i,j+1}}\boldsymbol{\Gamma}_{\boldsymbol{HJ}}^{i:j+1}\right\|_F^2\left\|\boldsymbol{\Gamma}_{\boldsymbol{HJ}}^{i:j+1}\right\|^2 \leq Cn(i-j+1)$, and the fact that multiplying a matrix by $\boldsymbol{P}_{J_j}$ cannot increase its norm. Writing for concision in the subsequent expression

$$A = \left|\left\|\boldsymbol{\Gamma}_{\boldsymbol{HJ}}^{i:j+1}\boldsymbol{P}_{J_j}\tilde{\boldsymbol{W}}_{(:,1)}^j\right\|_2 - \mathbb{1}_{\mathcal{E}_H^{i,j+1}}\left\|\boldsymbol{\Gamma}_{\boldsymbol{HJ}}^{i:j+1}\boldsymbol{P}_{J_j}\tilde{\boldsymbol{W}}_{(:,1)}^j\right\|_2\right|$$

it follows that

$$\mathbb{P}\left[\left\|\mathbf{\Gamma}_{HJ}^{i:j+1}\mathbf{P}_{J_j}\tilde{\mathbf{W}}_{(:,1)}^j\right\|_2 > C''\right] \le \mathbb{P}\left[\mathbb{1}_{\mathcal{E}_H^{i,j+1}}\left\|\mathbf{\Gamma}_{HJ}^{i:j+1}\mathbf{P}_{J_j}\tilde{\mathbf{W}}_{(:,1)}^j\right\|_2 > C''\right] + \mathbb{P}\left[A > 0\right]$$

$$\le 2\exp\left(-c'\frac{n}{L}\right) + C\exp\left(-c''\frac{n}{L}\right) \le C'\exp\left(-c\frac{n}{L}\right)$$

for appropriate constants, where we used D.18. Since on $\mathcal{E}_{\delta K \rho}$ we have

$$\left\|\mathbf{P}_{S^{j\perp}}\mathbf{Q}^j(\mathbf{x})\frac{\boldsymbol{\rho}^{j-1}(\mathbf{x})}{\|\boldsymbol{\alpha}^{j-2}(\mathbf{x})\|_2}\right\|_2 \le \left\|\mathbf{Q}^j(\mathbf{x})\frac{\boldsymbol{\rho}^{j-1}(\mathbf{x})}{\|\boldsymbol{\alpha}^{j-2}(\mathbf{x})\|_2}\right\|_2 \le 2K_s,$$

we obtain

$$\mathbb{P}\left[\mathbb{1}_{\mathcal{E}_{\delta K \rho}}\left\|\mathbf{\Gamma}_{HJ}^{i:j}\mathbf{P}_{J_{j-1}}\frac{\boldsymbol{\rho}^{j-1}(\mathbf{x})}{\|\boldsymbol{\alpha}^{j-2}(\mathbf{x})\|_2}\right\|_2 > 2C''K_s\right] \le C'\exp\left(-c\frac{n}{L}\right) \tag{D.95}$$

hence

$$\mathbb{P}\left[\tilde{a}_k > K_s\right] \le C\exp\left(-c'\frac{n}{L}\right) \tag{D.96}$$

for appropriate constants.

We now turn to controlling the $\tilde{b}_{qij}$. Note that

$$\mathbb{1}_{\mathcal{E}_{\delta K \rho}}\left|\prod_{k=i}^q s_k\right| = \mathbb{1}_{\mathcal{E}_{\delta K \rho}}\frac{\|\boldsymbol{\alpha}^q(\mathbf{x})\|_2}{\|\boldsymbol{\alpha}^{i-1}(\mathbf{x})\|_2}\left|\prod_{k=i}^q\left(1 + \frac{\boldsymbol{\alpha}^k(\mathbf{x})^*}{\|\boldsymbol{\alpha}^k(\mathbf{x})\|_2^2}\mathbf{Q}^k(\mathbf{x})\mathbf{W}^k\boldsymbol{\alpha}^{k-1}(\mathbf{x})\right)\right| \tag{D.97}$$

$$\le 3\left(1 + 2K_{\mathcal{J}}\right)^{q-i} \le 3e^{2K_{\mathcal{J}}L} \le 9 \ a.s.$$

where in the last inequality we used $2K_{\mathcal{J}}L < 1$. Additionally, we have

$$\mathbb{1}_{\mathcal{E}_{\delta K \rho}}\frac{\boldsymbol{\alpha}^i(\mathbf{x})^*}{\|\boldsymbol{\alpha}^i(\mathbf{x})^*\|_2}\mathbf{\Gamma}_{HJ}^{i-1:j+1}\mathbf{P}_{J_j}\frac{\boldsymbol{\rho}^j(\mathbf{x})}{\|\boldsymbol{\alpha}^j(\mathbf{x})\|_2}$$

$$\stackrel{d}{=} \mathbb{1}_{\mathcal{E}_{\delta K \rho}}\frac{\boldsymbol{\alpha}^i(\mathbf{x})^*\mathbf{P}_{J_{i-1}}}{\|\boldsymbol{\alpha}^i(\mathbf{x})^*\|_2}\tilde{\mathbf{W}}^{i-1}\mathbf{P}_{S^{i-2\perp}}\mathbf{\Gamma}_{HJ}^{i-2:j+1}\mathbf{P}_{J_j}\frac{\boldsymbol{\rho}^j(\mathbf{x})}{\|\boldsymbol{\alpha}^j(\mathbf{x})\|_2}$$

$$\stackrel{.}{=} \mathbb{1}_{\mathcal{E}_{\delta K \rho}}\frac{\boldsymbol{\alpha}^i(\mathbf{x})^*\mathbf{P}_{J_{i-1}}}{\|\boldsymbol{\alpha}^i(\mathbf{x})^*\|_2}\tilde{\mathbf{W}}^{i-1}\mathbf{u} \stackrel{d}{=} \sigma g$$

where $\tilde{\mathbf{W}}^{i-1}$ is an copy of $\mathbf{W}^{i-1}$ that is independent of all the other variables, we defined

$$\mathbf{u} = \mathbf{P}_{S^{i-2\perp}}\mathbf{\Gamma}_{HJ}^{i-2:j+1}\mathbf{P}_{J_j}\frac{\boldsymbol{\rho}^j(\mathbf{x})}{\|\boldsymbol{\alpha}^j(\mathbf{x})\|_2}$$

and $g$ is a standard normal variable. In the above expression,

$$\sigma^2 = \mathbb{1}_{\mathcal{E}_{\delta K \rho}}\frac{2}{n}\left\|\boldsymbol{\alpha}^i(\mathbf{x})^*\mathbf{P}_{J_{i-1}}/\left\|\boldsymbol{\alpha}^i(\mathbf{x})^*\right\|_2\right\|_2^2\|\mathbf{u}\|_2^2 \le \mathbb{1}_{\mathcal{E}_{\delta K \rho}}\frac{2\|\mathbf{u}\|_2^2}{n}.$$

Note also that $\mathbf{\Gamma}_{HJ}^{i-2:j+1}$ is well-defined since $i \ge j - 1$.

We therefore have

$$\mathbb{P}\left[\left|\mathbb{1}_{\mathcal{E}_{\delta K \rho}}\frac{\boldsymbol{\alpha}^i(\mathbf{x})^*}{\|\boldsymbol{\alpha}^i(\mathbf{x})^*\|_2}\mathbf{\Gamma}_{HJ}^{i-1:j+1}\mathbf{P}_{J_j}\frac{\boldsymbol{\rho}^j(\mathbf{x})}{\|\boldsymbol{\alpha}^j(\mathbf{x})\|_2}\right| > \frac{CK_s}{\sqrt{L}}\right]$$

$$\le \mathbb{P}\left[\mathbb{1}_{\mathcal{E}_{\delta K \rho}}\|\mathbf{u}\|_2 > K_s\right] + \mathbb{P}\left[\left|g\sqrt{\frac{2}{n}}K_s\right| > \frac{K_s}{\sqrt{L}}\right]$$

$$\le C'e^{-c\frac{n}{L}} + 2e^{-c'\frac{n}{L}} \le C''e^{-c\frac{n}{L}} \tag{D.98}$$

where we used (D.95) and the Gaussian tail probability to bound the first and second terms in the second line respectively. Combining the above with (D.97) we obtain

$$\mathbb{P}\left[\left|\tilde{b}_{qij}\right| > \frac{K_s}{\sqrt{L}}\right] \le C''e^{-c\frac{n}{L}}. \tag{D.99}$$

We now turn to controlling $\tilde{c}_{ji}^p$. If $i \geq \ell'$ we have

$$
\begin{aligned}
\tilde{c}_{j,i+1}^p &= \mathbb{1}_{\mathcal{E}_{\delta K \rho}} \frac{\prod\limits_{k=i+2}^{j-1} s_k \boldsymbol{\alpha}^{i+1}(\boldsymbol{x})^*}{\left\|\boldsymbol{\alpha}^{i+1}(\boldsymbol{x})^*\right\|_2} \boldsymbol{\Gamma}_{\boldsymbol{HJ}}^{i:\ell'} \boldsymbol{v}_p \\
&\overset{d}{=} \mathbb{1}_{\mathcal{E}_{\delta K \rho}} \frac{\prod\limits_{k=i+2}^{j-1} s_k \boldsymbol{\alpha}^{i+1}(\boldsymbol{x})^* \boldsymbol{P}_{J_i}}{\left\|\boldsymbol{\alpha}^{i+1}(\boldsymbol{x})^*\right\|_2} \tilde{\boldsymbol{W}}^i \boldsymbol{P}_{S^{i-1}\perp} \boldsymbol{\Gamma}_{\boldsymbol{HJ}}^{i-1:\ell'} \boldsymbol{v}_p \overset{d}{=} \sigma g
\end{aligned}
\tag{D.100}
$$

where $g$ is a standard normal and

$$
\sigma^2 = \mathbb{1}_{\mathcal{E}_{\delta K \rho}} \frac{2}{n} \left(\prod_{k=i+2}^{j-1} s_k\right)^2 \left\|\boldsymbol{P}_{S^{i-1}\perp} \boldsymbol{\Gamma}_{\boldsymbol{HJ}}^{i-1:\ell'} \boldsymbol{v}_p\right\|_2^2 \leq \mathbb{1}_{\mathcal{E}_{\delta K \rho}} \frac{2C}{n} \left\|\boldsymbol{\Gamma}_{\boldsymbol{HJ}}^{i-1:\ell'} \boldsymbol{v}_p\right\|_2^2 \ a.s.
$$

for some constant $C$ where we used (D.97). We also have $\left\|\boldsymbol{\Gamma}_{\boldsymbol{HJ}}^{i-1:\ell'} \boldsymbol{v}_p\right\|_2^2 \leq \tilde{C}$ on $\mathcal{E}_H^{i-1,\ell'}$. Lemma D.18 and a Gaussian tail bound then give

$$
\mathbb{P}\left[\left|\tilde{c}_{j,i+1}^p\right| > \sqrt{\frac{d}{n}}\right] \leq
\begin{array}{l}
\mathbb{P}\left[\mathbb{1}_{\mathcal{E}_{\delta K \rho} \cap \mathcal{E}_H^{i-1,\ell'}} \left|\prod\limits_{k=i+2}^{j-1} s_k \frac{\boldsymbol{\alpha}^{i+1}(\boldsymbol{x})^* \boldsymbol{\Gamma}_{\boldsymbol{HJ}}^{i:\ell'} \boldsymbol{v}_p}{\|\boldsymbol{\alpha}^{i+1}(\boldsymbol{x})^*\|_2}\right| > \sqrt{\frac{d}{n}}\right] \\
+ \mathbb{P}\left[\left(\mathbb{1}_{\mathcal{E}_{\delta K \rho}} - \mathbb{1}_{\mathcal{E}_{\delta K \rho} \cap \mathcal{E}_H^{i-1,\ell'}}\right) \left|\prod\limits_{k=i+2}^{j-1} s_k \frac{\boldsymbol{\alpha}^{i+1}(\boldsymbol{x})^* \boldsymbol{\Gamma}_{\boldsymbol{HJ}}^{i:\ell'} \boldsymbol{v}_p}{\|\boldsymbol{\alpha}^{i+1}(\boldsymbol{x})^*\|_2}\right| > 0\right]
\end{array}
$$

$$
\leq \mathbb{P}\left[\sqrt{\frac{2}{n}}\tilde{C}g > \sqrt{\frac{d}{n}}\right] + \mathbb{P}\left[\left(\mathcal{E}_H^{i-1,\ell'}\right)^c\right] \leq 2e^{-cd} + Ce^{-c'\frac{n}{L}}
\tag{D.101}
$$

for appropriate constants.

Additionally, if $i = \ell' - 1$ we have from (D.97) for some fixed $\boldsymbol{v}_p = \boldsymbol{v}_f$

$$
\begin{aligned}
\left|\tilde{c}_{j,\ell'}^f\right| &= \left|\mathbb{1}_{\mathcal{E}_{\delta K \rho}} \prod_{k=\ell'+1}^{j-1} s_k \frac{\boldsymbol{\alpha}^{\ell'}(\boldsymbol{x})^*}{\left\|\boldsymbol{\alpha}^{\ell'}(\boldsymbol{x})\right\|_2} \boldsymbol{\Gamma}_{\boldsymbol{HJ}}^{\ell'-1:\ell'} \boldsymbol{v}_f\right| \\
&= \left|\mathbb{1}_{\mathcal{E}_{\delta K \rho}} \prod_{k=\ell'+1}^{j-1} s_k \frac{\boldsymbol{\alpha}^{\ell'}(\boldsymbol{x})^*}{\left\|\boldsymbol{\alpha}^{\ell'}(\boldsymbol{x})\right\|_2} \boldsymbol{v}_f\right| \\
&= \left|\mathbb{1}_{\mathcal{E}_{\delta K \rho}} \prod_{k=\ell'+1}^{j-1} s_k\right| \leq 9 \ a.s.
\end{aligned}
\tag{D.102}
$$

If however $\boldsymbol{v}_p = \boldsymbol{v}_u$ is drawn from $\mathrm{Unif}(\mathbb{S}^{n-1})$, if we denote by $\boldsymbol{g}$ a vector with independent standard Gaussian entries we have

$$
\mathbb{1}_{\mathcal{E}_{\delta K \rho}} \frac{\boldsymbol{\alpha}^{\ell'}(\boldsymbol{x})^*}{\left\|\boldsymbol{\alpha}^{\ell'}(\boldsymbol{x})\right\|_2} \boldsymbol{v}_u \overset{d}{=} \widehat{\boldsymbol{e}}_1^* \boldsymbol{v}_u \overset{d}{=} \frac{g_1}{\|\boldsymbol{g}\|_2}.
$$

From Bernstein's inequality it follows that $\mathbb{P}\left[\|\boldsymbol{g}\|_2^2 < \frac{n}{2}\right] \leq e^{-cn}$. Combining this with a Gaussian tail bound gives

$$
\mathbb{P}\left[\left|\frac{g_1}{\|\boldsymbol{g}\|_2}\right| > \sqrt{\frac{d}{n}}\right] \leq \mathbb{P}\left[\|\boldsymbol{g}\|_2^2 < \frac{n}{2}\right] + \mathbb{P}\left[|g_1| > \sqrt{\frac{d}{2}}\right] \leq e^{-cn} + 2e^{-c'd}
$$

for some constants $c, c'$. From (D.97) it follows that

$$
\mathbb{P}\left[\left|\tilde{c}_{j,\ell'}^u\right| > \sqrt{\frac{d}{n}}\right] = \mathbb{P}\left[\left|\mathbb{1}_{\mathcal{E}_{\delta K \rho}} \prod_{k=\ell'+1}^{j-1} s_k \frac{\boldsymbol{\alpha}^{\ell'}(\boldsymbol{x})^*}{\left\|\boldsymbol{\alpha}^{\ell'}(\boldsymbol{x})\right\|_2} \boldsymbol{v}_u\right| > \sqrt{\frac{d}{n}}\right] \leq e^{-cn} + 2e^{-c'd}
$$

for appropriate constants. $\qquad \square$

*Proof of lemma D.16.* **Part (i).** We denote the set of all such terms with $r$ G-chains by $\overleftarrow{G}_{r,p}$. Considering first the contribution from the terms with a single G-chain, denoted $\overleftarrow{G}_{1p}$, we have

$$\sum_{\overleftarrow{g}^{1,p} \in \overleftarrow{G}_{1p}} \overleftarrow{g}^{1,p} = \tilde{a}_\ell \sum_{j=\ell'+1}^{\ell} \tilde{c}_{\ell,j}^p$$

where

$$\sum_{j=\ell'+1}^{\ell} \tilde{c}_{\ell,j}^p = \sum_{j=\ell'+1}^{\ell} \mathbb{1}_{\mathcal{E}_{\delta K_\rho}} \frac{\prod_{k=j+1}^{\ell-1} s_k \boldsymbol{\alpha}^j(\boldsymbol{x})^* \boldsymbol{\Gamma}_{\boldsymbol{H}\mathcal{J}}^{j-1:\ell'} \boldsymbol{v}^p}{\|\boldsymbol{\alpha}^j(\boldsymbol{x})\|_2}.$$

Denoting by $\sigma(A_1, \ldots, A_k)$ the sigma-algebra generated by a the random variables $A_1, \ldots, A_k$, we define a filtration

$$\begin{aligned} \mathcal{F}^{\ell'-1} &= \sigma\left(\boldsymbol{v}_p, \boldsymbol{\rho}^1(\boldsymbol{x}), \ldots, \boldsymbol{\rho}^L(\boldsymbol{x})\right), \\ \mathcal{F}^j &= \sigma\left(\boldsymbol{v}_p, \boldsymbol{\rho}^1(\boldsymbol{x}), \ldots, \boldsymbol{\rho}^L(\boldsymbol{x}), \boldsymbol{H}^{\ell'}, \ldots, \boldsymbol{H}^j\right), \; j = \ell', \ldots, \ell. \end{aligned} \tag{D.103}$$

The sequence $\{X_i\} = \left\{ \sum_{j=\ell'+1}^{1+i} \tilde{c}_{\ell,j} \right\}$ is adapted to the filtration, and since the summands are linear in the zero mean $\{\boldsymbol{H}^k\}$ the sequence is a martingale ($\mathbb{E}\, X_{i+1}|\mathcal{F}^i = X_i$). The martingale difference sequence is

$$\Delta_i = X_i - X_{i-1} = \tilde{c}_{\ell,i+1}^p = \mathbb{1}_{\mathcal{E}_{\delta K_\rho}} \frac{\prod_{k=i+2}^{\ell-1} s_k \boldsymbol{\alpha}^{i+1}(\boldsymbol{x})^* \boldsymbol{\Gamma}_{\boldsymbol{H}\mathcal{J}}^{i:\ell'} \boldsymbol{v}_p}{\|\boldsymbol{\alpha}^{i+1}(\boldsymbol{x})\|_2}$$

giving

$$\sum_{j=\ell'+1}^{\ell} \tilde{c}_{\ell,j}^p = X_{\ell-1} = \sum_{j=\ell'+1}^{\ell-1} \Delta_i + X_{\ell'} = \sum_{j=\ell'+1}^{\ell-1} \Delta_i + \tilde{c}_{\ell,\ell'+1}^p.$$

We cannot control this sum directly because we do not have almost sure control of the martingale differences. To remedy this, we recall the event $\mathcal{E}_H^{i-1,\ell'+1}$ defined in lemma D.18, and decompose the sum of interest into

$$\left| \sum_{j=\ell'+1}^{\ell-1} \Delta_i \right| \le \left| \sum_{j=\ell'+1}^{\ell-1} \Delta_i - \mathbb{1}_{\mathcal{E}_H^{i-1,\ell'+1}} \Delta_i \right| + \left| \sum_{j=\ell'+1}^{\ell-1} \mathbb{1}_{\mathcal{E}_H^{i-1,\ell'+1}} \Delta_i \right|. \tag{D.104}$$

Notice that the second sum is also a sum of zero-mean martingale differences. Using (D.100), we have

$$\mathbb{1}_{\mathcal{E}_H^{i-1,\ell'+1}} \Delta_i \stackrel{d}{=} \mathbb{1}_{\mathcal{E}_H^{i-1,\ell'+1}} \sigma g$$

where $g \sim \mathcal{N}(0,1)$ and $\mathbb{1}_{\mathcal{E}_H^{i-1,\ell'+1}} \sigma^2 = \mathbb{1}_{\mathcal{E}_H^{i-1,\ell'+1}} \frac{2}{n} \left( \prod_{k=i+1}^{\ell-1} s_k \right)^2 \left\| \boldsymbol{\Gamma}_{\boldsymbol{H}\mathcal{J}}^{i-1:\ell'} \boldsymbol{v}_p \right\|_2^2 \le \frac{C}{n}$ almost surely

for some constant $C$. It follows that

$$\mathbb{E}\left[ \exp\left( \lambda \mathbb{1}_{\mathcal{E}_H^{i-1,\ell'+1}} \Delta_i \right) \Big| \mathcal{F}^{i-1} \right] \le \exp\left( cn\lambda^2 \right) \; \forall \lambda, a.s.$$

and we can apply Freedman's inequality for martingales with sub-Gaussian increments (lemma G.7) to conclude that for some $d \ge 0$

$$\mathbb{P}\left[ \left| \sum_{j=\ell'}^{\ell-1} \mathbb{1}_{\mathcal{E}_H^{i-1,\ell'+1}} \Delta_i \right| > \sqrt{d} \right] \le 2 \exp\left( -c \frac{dn}{L} \right).$$

As for the first term in (D.104), using lemma D.18 we have

$$\mathbb{P}\left[ \left| \sum_{j=\ell'}^{\ell-1} \Delta_i - \mathbb{1}_{\mathcal{E}_H^{i-1,\ell'+1}} \Delta_i \right| > 0 \right] \le \sum_{i=1}^{\ell-\ell'} \mathbb{P}\left[ \left( \mathcal{E}_H^{i-1,\ell'+1} \right)^c \right] \le LC' e^{-c \frac{n}{L}} \le C' e^{-c' \frac{n}{L}}$$

for appropriate constants, where we assumed $n \geq KL \log L$ for some $K$. Combining the above with (D.94), and using (D.101) to give $\mathbb{P}\left[\left|\tilde{a}_\ell \tilde{c}^p_{\ell,\ell'+1}\right| > \tilde{C}\right] \leq C' e^{-c\frac{n}{L}}$ for some constants and applying the triangle inequality, we have

$$\mathbb{P}\left[\left|\sum_{\overleftarrow{g}^{1,p} \in \overleftarrow{G}_{1p}} \overleftarrow{g}^{1,p}\right| > C\sqrt{d}\right] \leq \mathbb{P}\left[\left|\tilde{a}_\ell \tilde{c}^p_{\ell,\ell'+1}\right| + |\tilde{a}_\ell|\left|\sum_{j=\ell'+1}^{\ell-1} \Delta_i\right| > C\sqrt{d}\right] \tag{D.105}$$
$$\leq C' e^{-c\frac{n}{L}} + C'' e^{-c'\frac{dn}{L}}$$

for appropriate constants.

Having controlled the sum of terms in $\overleftarrow{G}_{1p}$, we next consider a sum over the terms in $\overleftarrow{G}_{r,p}$ for $r \geq 2$. The argument will be very similar to the $\overleftarrow{G}_{1p}$ case, with some additional technical details.

Note that since different G-chains must be separated by an $\boldsymbol{H}^i$ matrix for some $i$ and we consider only terms with $i_1 \geq \ell' + 1$, the minimal starting index of the $r$-th chain (indexed by $j$ below for clarity) is $\ell' + 1 + 2(r-1)$. The sum of all possible terms is thus

$$\sum_{\overleftarrow{g}^{r,p} \in \overleftarrow{G}_{r,p}} \overleftarrow{g}^{r,p} = \tilde{a}_\ell \sum_{\substack{\ell'+2r-1 \leq j \leq \ell, \\ (i_1,\ldots,i_{2r-2}) \in \mathcal{C}_{r-1}(\ell'+1, j-2)}} \tilde{b}_{\ell,j,i_{2r-2}} \prod_{m=1}^{r-2} \tilde{b}_{i_{2m+2},i_{2m+1},i_{2m}} \tilde{c}^p_{i_2,i_1}$$
$$\doteq \sum_{j=\ell'+2r-1}^{\ell} \tilde{p}^r_{\ell,j}$$

The constraints on the indices $i_1,\ldots,i_{2r-2}$ are similar to those in (D.84), with the starting index reflecting the constraint $i_1 > \ell'$ in the definition of $\overleftarrow{G}_{r,p}$. We once again define the filtration $\mathcal{F}^j$ as in (D.103) for $\ell - 1 \leq j \leq \ell$. Noting that $\tilde{a}_\ell, \tilde{b}_{k,l,m} = \mathbb{1}_{\mathcal{E}_{\delta K\rho}} \prod_{q=l+1}^{k-1} s_q \frac{\boldsymbol{\alpha}^l(\boldsymbol{x})^* \boldsymbol{\Gamma}^{l-1:m+1}_{\boldsymbol{HJ}} \boldsymbol{P}_{J_m} \boldsymbol{\rho}^m(\boldsymbol{x})}{\|\boldsymbol{\alpha}^l(\boldsymbol{x})\|_2 \|\boldsymbol{\alpha}^{m-1}(\boldsymbol{x})\|_2}$ and

$\tilde{c}^p_{k,l} = \mathbb{1}_{\mathcal{E}_{\delta K\rho}} \frac{\prod_{q=l+1}^{k-1} s_q \boldsymbol{\alpha}^l(\boldsymbol{x})^* \boldsymbol{\Gamma}^{l-1:\ell'}_{\boldsymbol{HJ}} \boldsymbol{v}_p}{\|\boldsymbol{\alpha}^l(\boldsymbol{x})\|_2}$ are all $\mathcal{F}^{l-1}$-measurable, the index constraints imply that

$$X^r_i = \sum_{j=\ell'+2r-1}^{i+1} \tilde{p}^r_{\ell,j}$$

is $\mathcal{F}^i$-measurable and thus the sequence $\{X^r_i\}$ is adapted to the filtration. $\tilde{b}_{\ell,i+1,i_{2r-2}}$ is a linear function of the zero-mean variables $\boldsymbol{H}^i$ for any choice of $i_{2r-2}$, and we can replace $\boldsymbol{H}^i$ with $\tilde{\boldsymbol{W}}^i \boldsymbol{P}_{S^{i-1}\perp}$ where $\tilde{\boldsymbol{W}}^i$ is an independent copy of $\boldsymbol{W}^i$ without altering the distribution of $X^r_i$. Since $\tilde{b}_{klm}$ for $k \leq i_{2r-2}$ is independent of the $\tilde{\boldsymbol{W}}^i$ for any choice of $l, m$, it follows that $\tilde{p}^r_{\ell,i+1}$ is also a linear function of the variables in $\tilde{\boldsymbol{W}}^i$ which have zero mean. Consequently

$$\mathbb{E}\, X^r_i | \mathcal{F}^{i-1} = \sum_{j=\ell'+2r-1}^{i} \tilde{p}^r_{\ell,j} = X^r_{i-1},$$

hence $\{X_i\}$ is a martingale sequence. Defining martingale differences

$$\Delta^r_i = X^r_i - X^r_{i-1} = \tilde{p}^r_{\ell,i+1}$$

we have

$$\sum_{j=\ell'+2r-1}^{\ell} \tilde{p}^r_{\ell,j} = \sum_{i=\ell'+2r-2}^{\ell-1} \Delta^r_i. \tag{D.106}$$

We define an event $\mathcal{E}^i_\Delta \in \mathcal{F}^i$ by

$$\mathcal{E}^i_\Delta = \begin{array}{c} \{|\tilde{a}_\ell| \leq C\} \cap \bigcap_{i_1+2 \leq i_2 \leq i_3 \leq i} \left\{\left|\tilde{b}_{i_3 i_2 i_1}\right| \leq \frac{K_s}{\sqrt{L}}\right\} \cap \bigcap_{i_1 \leq i_2 \leq i} \left\{\left|\tilde{c}^p_{i_2 i_1}\right| \leq C\sqrt{\frac{d}{n}}\right\} \\ \cap \bigcap_{i_1 \leq i} \left\{\mathbb{1}_{\mathcal{E}_{\delta K\rho}}\left\|\boldsymbol{\Gamma}^{i:i_1}_{\boldsymbol{HJ}} \boldsymbol{P}_{J^{i_1-1}} \frac{\boldsymbol{\rho}^{i_1-1}(\boldsymbol{x})}{\|\boldsymbol{\alpha}^{i_1-2}(\boldsymbol{x})\|_2}\right\|_2 \leq CK_s\right\} \end{array} \tag{D.107}$$

for $i_1 \geq \ell' + 1$ and decompose the sum in (D.106) into

$$\left| \sum_{i=\ell'+2r-2}^{\ell-1} \Delta_i^r \right| \leq \left| \sum_{i=\ell'+2r-2}^{\ell-1} \Delta_i^r - \mathbb{1}_{\mathcal{E}_\Delta^{i-1}} \Delta_i^r \right| + \left| \sum_{i=\ell'+2r-2}^{\ell-1} \mathbb{1}_{\mathcal{E}_\Delta^{i-1}} \Delta_i^r \right|. \tag{D.108}$$

In order to control the second term, we note that

$$\mathbb{1}_{\mathcal{E}_\Delta^{i-1}} \Delta_i^r = \mathbb{1}_{\mathcal{E}_\Delta^{i-1}} \tilde{p}_{\ell,i+1}^r$$

$$= \mathbb{1}_{\mathcal{E}_\Delta^{i-1}} \tilde{a}_\ell \sum_{(i_1,\ldots,i_{2r-2}) \in \mathcal{C}_{r-1}(\ell'+1, i-1)} \tilde{b}_{\ell,i+1,i_{2r-2}} \prod_{m=1}^{r-2} \tilde{b}_{i_{2m+2}, i_{2m+1}, i_{2m}} \tilde{c}_{i_2,i_1}^p$$

$$= \mathbb{1}_{\mathcal{E}_\Delta^{i-1}} \tilde{a}_\ell \sum_{(i_1,\ldots,i_{2r-2}) \in \mathcal{C}_{r-1}(\ell'+1, i-1)} \mathbb{1}_{\mathcal{E}_{\delta K \rho}} \prod_{q=i+2}^{\ell-1} s_q \frac{\boldsymbol{\alpha}^{i+1}(\boldsymbol{x})^* \boldsymbol{\Gamma}_{\boldsymbol{HJ}}^{i:i_{2r-2}+1} \boldsymbol{P}_{J^{i_{2r-2}}} \boldsymbol{\rho}^{i_{2r-2}}(\boldsymbol{x})}{\|\boldsymbol{\alpha}^{i+1}(\boldsymbol{x})\|_2 \|\boldsymbol{\alpha}^{i_{2r-2}-1}(\boldsymbol{x})\|_2} \\ * \prod_{m=1}^{r-2} \tilde{b}_{i_{2m+2}, i_{2m+1}, i_{2m}} \tilde{c}_{i_2,i_1}^p$$

and using $\boldsymbol{\Gamma}_{\boldsymbol{HJ}}^{i:i_{2r-2}+1} = \boldsymbol{P}_{J_i} \boldsymbol{H}^i \boldsymbol{\Gamma}_{\boldsymbol{HJ}}^{i-1:i_{2r-2}+1} \overset{d}{=} \boldsymbol{P}_{J_{(i)}} \tilde{\boldsymbol{W}}^i \boldsymbol{P}_{S^{i-1}\perp} \boldsymbol{\Gamma}_{\boldsymbol{HJ}}^{i-1:i_{2r-2}+1}$ where $\tilde{\boldsymbol{W}}^i$ is an independent copy of $\boldsymbol{W}^i$ gives

$$\mathbb{1}_{\mathcal{E}_\Delta^{i-1}} \Delta_i^r \overset{d}{=} \sigma g$$

where $g \sim \mathcal{N}(0,1)$ and

$$\sigma = \sqrt{\frac{2}{n}} \mathbb{1}_{\mathcal{E}_{\delta K \rho} \cap \mathcal{E}_\Delta^{i-1}} \left| \prod_{q=i+2}^{\ell-1} s_q \right| \tilde{a}_\ell \left\| \sum_{\substack{(i_1,\ldots,i_{2r-2}) \\ \in \mathcal{C}_{r-1}(\ell'+1, i-1)}} \frac{\boldsymbol{P}_{S^{i-1}\perp} \boldsymbol{\Gamma}_{\boldsymbol{HJ}}^{i-1:i_{2r-2}+1} \boldsymbol{P}_{J^{i_{2r-2}}} \boldsymbol{\rho}^{i_{2r-2}}(\boldsymbol{x})}{\|\boldsymbol{\alpha}^{i_{2r-2}-1}(\boldsymbol{x})\|_2} \\ * \prod_{m=1}^{r-2} \tilde{b}_{i_{2m+2}, i_{2m+1}, i_{2m}} \tilde{c}_{i_2,i_1}^p \right\|_2$$

$$\leq \sqrt{\frac{2}{n}} \mathbb{1}_{\mathcal{E}_{\delta K \rho} \cap \mathcal{E}_\Delta^{i-1}} \left| \prod_{q=i+2}^{\ell-1} s_q \right| \tilde{a}_\ell \sum_{\substack{(i_1,\ldots,i_{2r-2}) \\ \in \mathcal{C}_{r-1}(\ell'+1, i-1)}} \frac{\left\| \boldsymbol{\Gamma}_{\boldsymbol{HJ}}^{i-1:i_{2r-2}+1} \boldsymbol{P}_{J^{i_{2r-2}}} \boldsymbol{\rho}^{i_{2r-2}}(\boldsymbol{x}) \right\|_2}{\|\boldsymbol{\alpha}^{i_{2r-2}-1}(\boldsymbol{x})\|_2} \\ * \left| \prod_{m=1}^{r-2} \tilde{b}_{i_{2m+2}, i_{2m+1}, i_{2m}} \tilde{c}_{i_2,i_1}^p \right|$$

$$\leq \sqrt{\frac{2}{n}} \mathbb{1}_{\mathcal{E}_{\delta K \rho} \cap \mathcal{E}_\Delta^{i-1}} \left| \prod_{q=i+2}^{\ell-1} s_q \right| \tilde{a}_\ell L^{2r-2} \max_{i_1 \leq i-1} \mathbb{1}_{\mathcal{E}_{\delta K \rho} \cap \mathcal{E}_\Delta^{i-1}} \frac{\left\| \boldsymbol{\Gamma}_{\boldsymbol{HJ}}^{i-1:i_1} \boldsymbol{P}_{J^{i_1-1}} \boldsymbol{\rho}^{i_1-1}(\boldsymbol{x}) \right\|_2}{\|\boldsymbol{\alpha}^{i_1-2}(\boldsymbol{x})\|_2} \\ * \max_{i_1+2 \leq i_2 \leq i_3 \leq i-1} \left( \mathbb{1}_{\mathcal{E}_{\delta K \rho} \cap \mathcal{E}_\Delta^{i-1}} \left| \tilde{b}_{i_3 i_2 i_1} \right| \right)^{r-2} \max_{i_1 \leq i_2 \leq i-1} \mathbb{1}_{\mathcal{E}_{\delta K \rho} \cap \mathcal{E}_\Delta^{i-1}} \left| \tilde{c}_{i_2 i_1}^p \right|$$

$$\underset{a.s.}{\leq} C \frac{\sqrt{dL}}{n} L^{2r-2} \left( \frac{K_s}{\sqrt{L}} \right)^{r-1} = C \frac{\sqrt{dL}}{n} \left( L^{3/2} K_s \right)^{r-1}.$$

In the last inequality we used the definition of $\mathcal{E}_\Delta^i$ and the assumption $L^{3/2} K_s \leq 1$. It follows that

$$\mathbb{E}\left[ \exp\left( \lambda \mathbb{1}_{\mathcal{E}_\Delta^{i-1}} \Delta_i \right) \Big| \mathcal{F}^{i-1} \right] \leq \exp\left( \frac{cn^2\lambda^2}{dL \left( L^{3/2} K_s \right)^{2r-2}} \right) \quad \forall \lambda, a.s.$$

and we can apply Freedman's inequality for martingales with sub-Gaussian increments (lemma G.7) to conclude

$$\mathbb{P}\left[ \left| \sum_{i=\ell'+2r-2}^{\ell-1} \mathbb{1}_{\mathcal{E}_\Delta^{i-1}} \Delta_i \right| > \left( L^{3/2} K_s \right)^{r-1} \sqrt{\frac{dL}{n}} \right] \leq 2\exp\left( -c\frac{n}{L} \right). \tag{D.109}$$

It remains to bound the first term in (D.108). Using lemma D.15, (D.95) and taking a union bound over $i_1, i_2, i_3$ in (D.107) we have

$$\mathbb{P}\left[ \mathcal{E}_\Delta^i \right] \geq 1 - CL^3 e^{-c\frac{n}{L}} - C'L^2 \left( e^{-c\frac{n}{L}} + e^{-c'd} \right) \geq 1 - C'' e^{-c''\frac{n}{L}} - C''' e^{-c'''d}$$

where we assume $n \geq KL \log L$ and $d \geq K' \log L$ for some $K, K'$. An additional union bound over $i$ gives

$$\mathbb{P}\left[\left|\sum_{i=\ell'+2r-2}^{\ell-1} \Delta_i^r - \mathbb{1}_{\mathcal{E}_\Delta^{i-1}}\Delta_i^r\right| > 0\right] \leq \sum_{i=1}^{\ell-\ell'} \mathbb{P}\left[\left(\mathcal{E}_\Delta^{i-1}\right)^c\right]$$

$$\leq LC'\left(e^{-c\frac{n}{L}} - e^{-c'd}\right) \leq C''\left(e^{-c''\frac{n}{L}} - e^{-c'''d}\right)$$

for appropriate constants. Combining the above with (D.109) and recalling (D.106) gives

$$\mathbb{P}\left[\left|\sum_{j=\ell'+2r-1}^{\ell} \tilde{p}_{\ell,j}^r\right| > \left(L^{3/2}K_s\right)^{r-1}\sqrt{\frac{dL}{n}}\right] \leq Ce^{-c\frac{n}{L}} + C'e^{-c'd}$$

for some constants. $L^{3/2}K_s \leq \frac{1}{2}$ implies

$$\sum_{r=2}^{\lceil(\ell-\ell')/2\rceil}\left(L^{3/2}K_s\right)^{r-1} = L^{3/2}K_s \sum_{r=0}^{\lceil(\ell-\ell')/2\rceil-2}\left(L^{3/2}K_s\right)^r$$

$$= L^{3/2}K_s\left(\frac{1-\left(L^{3/2}K_s\right)^{\lceil(\ell-\ell')/2\rceil-1}}{1-L^{3/2}K_s}\right) \leq 2.$$

A final union bound over $r$ and a rescaling of $d$ gives

$$\mathbb{P}\left[\left|\sum_{r=2}^{\lceil(\ell-\ell')/2\rceil}\sum_{\overleftarrow{g}^r \in \overleftarrow{G}_r}\overleftarrow{g}^r\right| > \sqrt{\frac{dL}{n}}\right] = \mathbb{P}\left[\left|\sum_{r=2}^{\lceil(\ell-\ell')/2\rceil}\sum_{j=\ell'+2r-1}^{\ell}\tilde{p}_{\ell,j}^r\right| > \sqrt{\frac{dL}{n}}\right]$$

$$\leq CLe^{-c\frac{n}{L}} + C'Le^{-c'd} \leq C''e^{-c''\frac{n}{L}} + C'''e^{-c'''d}$$

(D.110)

for appropriate constants, again assuming assume $n \geq KL \log L$ and $\bar{d} \geq K' \log L$ for some $K, K'$. Combining the above with equation (D.105) and worsening constants gives

$$\mathbb{P}\left[\left|\sum_{r=1}^{\lceil(\ell-\ell')/2\rceil}\sum_{\overleftarrow{g}^r \in \overleftarrow{G}_r}\overleftarrow{g}^r\right| > \sqrt{\frac{dL}{n}}\right] \leq Ce^{-c\frac{n}{L}} + C'e^{-c'd}$$

for appropriate constants.

**Part (ii).** We consider terms in the sets $\overrightarrow{G}_{r,p}$ (with $i_1 = \ell'$ and $i_{2r} \leq \ell - 1$). In contrast to the previous section, the bounds on these terms will differ based on the value of the $p$ subscript (denoting whether we use a fixed vector $\boldsymbol{v}_f$ or a random vector $\boldsymbol{v}_u$). We first consider $\overrightarrow{G}_{1,p}$, noting

$$\sum_{\overleftarrow{g}^{1,p} \in \overleftarrow{G}_{1p}}\overleftarrow{g}^{1,p} = \sum_{j=\ell'}^{\ell-1}\tilde{a}_j\tilde{c}_{j,\ell'}^p.$$

Lemma D.15 and a union bound give

$$\mathbb{P}\left[\bigcap_{j=\ell'}^{\ell-1}\tilde{a}_j \leq K_s \cap \bigcap_{j=\ell'}^{\ell-1}\left|\tilde{c}_{j,\ell'}^f\right| \leq C \cap \bigcap_{j=\ell'}^{\ell-1}\left|\tilde{c}_{j,\ell'}^u\right| \leq C\sqrt{\frac{d_0}{n}}\right]$$

$$\geq 1 - LC'e^{-c\frac{n}{L}} - L\left(2e^{-c'd_0} - e^{-c''n}\right)$$

$$\geq 1 - C''e^{-c'''\frac{n}{L}} - 2e^{-c''''d_0}$$

for appropriate constants, where we assume $n \geq KL \log L$ and $d_0 \geq K' \log L$ for some constants $K, K'$. With the same probability we have

$$\left| \sum_{\overleftarrow{g}^{1p} \in \overleftarrow{G}_{1p}} \overleftarrow{g}^{1p} \right| \leq \sum_{\overleftarrow{g}^{1p} \in \overleftarrow{G}_{1p}} |\overleftarrow{g}^{1p}| \leq CLK_s R_p^0 \leq CR_p^0 \tag{D.111}$$

where we defined

$$R_u^0 = \sqrt{\frac{d_0}{n}}, \ R_f^0 = 1$$

and used $LK_{\mathcal{J}} \leq 1$.

We next consider sums of terms in $\overrightarrow{G}_{r,p}$ for $r > 1$. In controlling the sum of these terms, the proof will proceed along similar lines to the previous section. The main tool we will be utilizing is martingale concentration. Recall that since $i_{2r} \leq \ell - 1$ and every two G-chains are separated by a matrix $\boldsymbol{H}^i$, the starting index of the final G-chain is no larger than $\ell - 2r + 1$. We thus have

$$\sum_{\overrightarrow{g}^{r,p} \in \overrightarrow{G}_{r,p}} \overrightarrow{g}^{r,p} = \sum_{\substack{\ell' \leq j \leq \ell - 2r + 1, \\ (i_3, \ldots, i_{2r}) \in \mathcal{C}_{r-1}(j+2, \ell-1)}} \tilde{a}_{i_{2r}} \prod_{m=2}^{r-1} \tilde{b}_{i_{2m+2}, i_{2m+1}, i_{2m}} \tilde{b}_{i_4, i_3, j} \tilde{c}_{j, \ell'}^p$$

$$\doteq \sum_{j=\ell'}^{\ell-2r+1} \tilde{p}_j^{r,p}.$$

Define a filtration

$$\mathcal{F}^0 = \sigma\left(\boldsymbol{v}_p, \boldsymbol{\rho}^1(\mathbf{x}), \ldots, \boldsymbol{\rho}^L(\mathbf{x})\right),$$
$$\mathcal{F}^j = \sigma\left(\boldsymbol{v}_p, \boldsymbol{\rho}^1(\mathbf{x}), \ldots, \boldsymbol{\rho}^L(\mathbf{x}), \boldsymbol{H}^\ell, \ldots, \boldsymbol{H}^{\ell-j+1}\right), \ j \in [\ell - \ell' + 1]$$

(note the reversed indexing convention compared to the filtration defined in (D.103)). Since $\tilde{p}_j^{r,p}$ is $\mathcal{F}^{\ell-j}$-measurable (as can be seen from (D.86)), we can define

$$X_i^{r,p} = \sum_{j=\ell-i}^{\ell} \tilde{p}_j^{r,p}$$

and it follows that $X_i^{r,p}$ is $\mathcal{F}^i$-measurable. Recalling from (D.86) that

$$\tilde{b}_{i_4, i_3, j} = \mathbb{1}_{\mathcal{E}_{\delta K \rho}} \prod_{k=i_3+1}^{i_4-1} s_k \frac{\boldsymbol{\alpha}^{i_3}(\boldsymbol{x})^* \boldsymbol{\Gamma}_{\boldsymbol{H}\mathcal{J}}^{i_3-1:j+1} \boldsymbol{P}_{\mathcal{J}j} \boldsymbol{\rho}^j(\boldsymbol{x})}{\|\boldsymbol{\alpha}^{i_3}(\boldsymbol{x})\|_2 \|\boldsymbol{\alpha}^{j-1}(\boldsymbol{x})\|_2}$$

and hence $\tilde{p}_j^{r,p}$ is linear in the zero-mean variables $\boldsymbol{H}^{j+1}$, we have

$$\mathbb{E} X_{i+1}^{r,p} | \mathcal{F}^i = \mathbb{E} \sum_{j=\ell-i-1}^{\ell} \tilde{p}_j^r | \mathcal{F}^i = \sum_{j=\ell-i}^{\ell} \tilde{p}_j^r + \underset{\boldsymbol{H}^1, \ldots, \boldsymbol{H}^{\ell-i}}{\mathbb{E}} \tilde{p}_{\ell-i-1}^r = \sum_{j=\ell-i}^{\ell} \tilde{p}_j^r = X_i$$

and thus the sequence $\{X_i^{r,p}\}$ is a martingale with respect to this filtration. Defining martingale differences $\Delta_i^{r,p} = X_i^{r,p} - X_{i-1}^{r,p} = \tilde{p}_{\ell-i}^{r,p}$ the sum of interest can be expressed as

$$\sum_{i=\ell'}^{\ell-2r+1} \tilde{p}_i^{r,p} = \sum_{i=2r-1}^{\ell-\ell'} \Delta_i^{r,p}. \tag{D.112}$$

We now define an event which we will shortly show holds with high probability:

$$\{|\tilde{a}_{i_{2r}}| \le K_s\}$$

$$\cap \bigcap_{m=2}^{r-1} \left\{ \left| \tilde{b}_{i_{2m+2},i_{2m+1},i_{2m}} \right| \le \frac{K_s}{\sqrt{L}} \right\}$$

$$\cap \left\{ \left| \tilde{c}_{\ell-i,\ell'}^f \right| \le C \right\}$$

$$\mathcal{E}_\Delta^{i-1} = \bigcap_{(i_3,\dots,i_{2r}) \in \mathcal{C}_{r-1}(\ell-i+2,\ell-1)} \left\{ \begin{array}{l} \cap \left\{ \left| \tilde{c}_{\ell-i,\ell'}^u \right| \le C \sqrt{\dfrac{d_1}{n}} \right\} \\[2ex] \cap \left\{ \mathbb{1}_{\mathcal{E}_{\delta K \rho}} \dfrac{\left\| \prod\limits_{k=i_3+1}^{i_4-1} s_k \boldsymbol{\alpha}^{i_3}(\boldsymbol{x})^* \boldsymbol{\Gamma}_{H\mathcal{J}}^{i_3-1:\ell-i+2} \boldsymbol{P}_{J_{\ell-i+1}} \right\|_2}{\left\| \boldsymbol{\alpha}^{i_3}(\boldsymbol{x}) \right\|_2} \le C\sqrt{L} \right\} \\[3ex] \cap \left\{ \mathbb{1}_{\mathcal{E}_{\delta K \rho}} \dfrac{\left\| \boldsymbol{P}_{S^{\ell-i\perp}} \boldsymbol{Q}_{J^{\ell-i}} \boldsymbol{\rho}^{\ell-i}(\boldsymbol{x}) \right\|_2}{\left\| \boldsymbol{\alpha}^{\ell-i-1}(\boldsymbol{x}) \right\|_2} \le 2K_s \right\} \end{array} \right\} .$$

$$\text{(D.113)}$$

Truncating the martingale difference on such an event gives

$$\mathbb{1}_{\mathcal{E}_\Delta^{i-1}} \Delta_i^{r,p}$$

$$= \mathbb{1}_{\mathcal{E}_\Delta^{i-1}} \tilde{p}_{\ell-i}^{r,p}$$

$$= \mathbb{1}_{\mathcal{E}_\Delta^{i-1}} \sum_{(i_3,\dots,i_{2r}) \in \mathcal{C}_{r-1}(\ell-i+2,\ell-1)} \tilde{a}_{i_{2r}} \prod_{m=2}^{r-1} \tilde{b}_{i_{2m+2},i_{2m+1},i_{2m}} \tilde{b}_{i_4,i_3,\ell-i} \tilde{c}_{\ell-i,\ell'}^p$$

$$= \mathbb{1}_{\mathcal{E}_\Delta^{i-1}}$$

$$\times \sum_{(i_3,\dots,i_{2r})} \tilde{a}_{i_{2r}} \prod_{m=2}^{r-1} \tilde{b}_{i_{2m+2},i_{2m+1},i_{2m}} \mathbb{1}_{\mathcal{E}_{\delta K \rho}} \prod_{k=i_3+1}^{i_4-1} s_k \frac{\boldsymbol{\alpha}^{i_3}(\boldsymbol{x})^* \boldsymbol{\Gamma}_{H\mathcal{J}}^{i_3-1:\ell-i+1} \boldsymbol{P}_{J^{\ell-i}} \boldsymbol{\rho}^{\ell-i}(\boldsymbol{x})}{\left\| \boldsymbol{\alpha}^{i_3}(\boldsymbol{x}) \right\|_2 \left\| \boldsymbol{\alpha}^{\ell-i-1}(\boldsymbol{x}) \right\|_2} \tilde{c}_{\ell-i,\ell'}^p$$

$$\overset{d}{=} \sigma^p g$$

for a standard normal $g$, where we used

$$\boldsymbol{\Gamma}_{H\mathcal{J}}^{i_3-1:\ell-i+1} \boldsymbol{P}_{J^{\ell-i}} \frac{\boldsymbol{\rho}^{\ell-i}(\boldsymbol{x})}{\left\| \boldsymbol{\alpha}^{\ell-i-1}(\boldsymbol{x}) \right\|_2}$$

$$= \boldsymbol{\Gamma}_{H\mathcal{J}}^{i_3-1:\ell-i+2} \boldsymbol{P}_{J_{\ell-i+1}} \boldsymbol{H}^{\ell-i+1} \boldsymbol{P}_{J^{\ell-i}} \frac{\boldsymbol{\rho}^{\ell-i}(\boldsymbol{x})}{\left\| \boldsymbol{\alpha}^{\ell-i-1}(\boldsymbol{x}) \right\|_2}$$

$$= \boldsymbol{\Gamma}_{H\mathcal{J}}^{i_3-1:\ell-i+2} \boldsymbol{P}_{J_{\ell-i+1}} \boldsymbol{H}^{\ell-i+1} \boldsymbol{Q}_{J^{\ell-i}} \frac{\boldsymbol{\rho}^{\ell-i}(\boldsymbol{x})}{\left\| \boldsymbol{\alpha}^{\ell-i-1}(\boldsymbol{x}) \right\|_2}$$

$$\overset{d}{=} \boldsymbol{\Gamma}_{H\mathcal{J}}^{i_3-1:\ell-i+2} \boldsymbol{P}_{J_{\ell-i+1}} \tilde{\boldsymbol{W}}^{\ell-i+1} \boldsymbol{P}_{S^{\ell-i\perp}} \boldsymbol{Q}_{J^{\ell-i}} \frac{\boldsymbol{\rho}^{\ell-i}(\boldsymbol{x})}{\left\| \boldsymbol{\alpha}^{\ell-i-1}(\boldsymbol{x}) \right\|_2}$$

with $\tilde{\boldsymbol{W}}^{\ell-i+1}$ an independent copy of $\boldsymbol{W}^{\ell-i+1}$ and we have defined

$$\sigma^p = \frac{\sqrt{\dfrac{2}{n}} \mathbb{1}_{\mathcal{E}_\Delta^{i-1} \cap \mathcal{E}_{\delta K \rho}} \left| \sum\limits_{(i_3,\dots,i_{2r}) \in \mathcal{C}_{r-1}(\ell-i+2,\ell-1)} \tilde{a}_{i_{2r}} \prod\limits_{m=2}^{r-1} \tilde{b}_{i_{2m+2},i_{2m+1},i_{2m}} \tilde{c}_{\ell-i,\ell'}^p \right|}{* \dfrac{\left\| \prod\limits_{k=i_3+1}^{i_4-1} s_k \boldsymbol{\alpha}^{i_3}(\boldsymbol{x})^* \boldsymbol{\Gamma}_{H\mathcal{J}}^{i_3-1:\ell-i+2} \boldsymbol{P}_{J_{\ell-i+1}} \right\|_2}{\left\| \boldsymbol{\alpha}^{i_3}(\boldsymbol{x}) \right\|_2} \dfrac{\left\| \boldsymbol{P}_{S^{\ell-i\perp}} \boldsymbol{Q}_{J^{\ell-i}} \boldsymbol{\rho}^{\ell-i}(\boldsymbol{x}) \right\|_2}{\left\| \boldsymbol{\alpha}^{\ell-i-1}(\boldsymbol{x}) \right\|_2}} .$$

Note that from (D.113), if we define

$$R_u = \sqrt{\frac{d_1}{n}}, R_f = 1,$$

the standard deviation $\sigma^p$ can be bounded as

$$\sigma^p \underset{a.s.}{\leq} \frac{CK_s^2}{\sqrt{n}} \left(\frac{K_s}{\sqrt{L}}\right)^{r-2} L^{2r-3/2} R_p \leq \frac{CR_p}{\sqrt{Ln}} \left(L^{3/2}K_s\right)^{r-1}$$

where in the first inequality we used a triangle inequality, bounded the number of summands by $L^{2r-2}$. In the second inequality we used $L^{3/2}K_s \leq \frac{1}{2}$.

Writing the sum in D.112 as

$$\left| \sum_{i=2r-1}^{\ell-\ell'} \Delta_i^{r,p} \right| \leq \left| \sum_{i=2r-1}^{\ell-\ell'} \left(1 - \mathbb{1}_{\mathcal{E}_\Delta^{i-1}}\right) \Delta_i^{r,p} \right| \leq \left| \sum_{i=2r-1}^{\ell-\ell'} \mathbb{1}_{\mathcal{E}_\Delta^{i-1}} \Delta_i^{r,p} \right|,$$

and recognizing that the second sum is over a zero-mean adapted sequence that obeys

$$\mathbb{E}\left[ \exp\left(\lambda \mathbb{1}_{\mathcal{E}_\Delta^{i-1}} \Delta_i \right) \Big| \mathcal{F}^{i-1} \right] \leq \exp\left( \frac{cn\lambda^2}{R_p^2 \left(L^{3/2}K_s\right)^{2r-2}} \right) \quad \forall \lambda, a.s.$$

an application of Freedman's inequality for martingales with sub-Gaussian increments (lemma G.7) gives

$$\mathbb{P}\left[ \left| \sum_{i=2r-1}^{\ell-\ell'} \mathbb{1}_{\mathcal{E}_\Delta^{i-1}} \Delta_i^{r,p} \right| > R_p \left(L^{3/2}K_s\right)^{r-1} \right] \leq 2\exp\left( \frac{-t^2}{2L\sigma_p^2} \right) \underset{a.s.}{\leq} 2e^{-cn}.$$

Turning now to controlling the probability of $\mathcal{E}_\Delta^{i-1}$ holding, we use lemmas D.15, D.18, the definition of $K_{\mathcal{J}}$ and a union bound to conclude

$$\mathbb{P}\left[ \left(\mathcal{E}_\Delta^{i-1}\right)^c \right] \leq L^3 Ce^{-c\frac{n}{L}} + L\left(2e^{-c'd_1} + e^{-c''n}\right) + L^2 C'e^{-c'''\frac{n}{L}} \leq C''e^{-c''''\frac{n}{L}} + e^{-c'd_1}$$

for appropriate constants, where we assumed $n \geq KL\log L, d_1 \geq K\log L$ for some $K, K'$. Combining the previous two results gives

$$\mathbb{P}\left[ \left| \sum_{i=\ell'}^{\ell-2r+1} \tilde{p}_i^{r,p} \right| > R_p \left(L^{3/2}K_s\right)^{r-1} \right] = \mathbb{P}\left[ \left| \sum_{i=2r-1}^{\ell-\ell'} \Delta_i^{r,p} \right| > R_p \left(L^{3/2}K_s\right)^{r-1} \right]$$

$$\leq \mathbb{P}\left[ \left| \sum_{i=2r-1}^{\ell-\ell'} \mathbb{1}_{\mathcal{E}_\Delta^{i-1}} \Delta_i^{r,p} \right| > R_p \left(L^{3/2}K_s\right)^{r-1} \right] + \mathbb{P}\left[ \left| \sum_{i=2r-1}^{\ell-\ell'} \left(1 - \mathbb{1}_{\mathcal{E}_\Delta^{i-1}}\right) \Delta_i^{r,p} \right| > 0 \right]$$

$$\leq 2e^{-cn} + L\mathbb{P}\left[ \left(\mathcal{E}_\Delta^{i-1}\right)^c \right] \leq Ce^{-c'\frac{n}{L}} + e^{-c''d_1}$$

for some constants. Noting as before that $\sum_{r=2}^{\lceil (\ell-\ell')/2 \rceil} \left(L^{3/2}K_s\right)^{r-1} \leq 2$, using (D.111) to bound the terms with $r = 1$, and setting $d_0 = d_1$ we obtain

$$\mathbb{P}\left[ \left| \sum_{r=2}^{\lceil (\ell-\ell')/2 \rceil} \sum_{\overrightarrow{g}^{r,p} \in \overrightarrow{G}_{r,p}} \overrightarrow{g}^{r,p} \right| > CR_p \right] \leq LC\left(e^{-c\frac{n}{L}} + e^{-c'd_1}\right) \leq C\left(e^{-c''\frac{n}{L}} + e^{-c'''d_1}\right)$$

for appropriate constants, where we used again $n \geq KL\log L, d_1 \geq K\log L$ for some $K, K'$. $\square$

**Lemma D.19.** *(Horn et al., 1994) Given a semidefinite matrix $\boldsymbol{A}$, for any partitioning*

$$\boldsymbol{A} = \begin{pmatrix} \boldsymbol{A}_{11} & \boldsymbol{A}_{12} & \dots & \boldsymbol{A}_{1b} \\ \boldsymbol{A}_{21} & \boldsymbol{A}_{22} & & \\ \vdots & & \ddots & \\ \boldsymbol{A}_{b1} & & & \boldsymbol{A}_{bb} \end{pmatrix}$$

*we have $\|\boldsymbol{A}\| \leq \sum_{i=1}^{b} \|\boldsymbol{A}_{ii}\|$.*

**Lemma D.20.** *Given a semidefinite matrix $\boldsymbol{A}$ and unit norm $\boldsymbol{v}$, if*

$$\mathbb{P}\left[\boldsymbol{v}^* \boldsymbol{A} \boldsymbol{v} \leq C'\right] \geq 1 - C\ell^p \exp\left(-c_1 \frac{n}{\ell}\right)$$

*and $n > \frac{2\log(9)\ell}{c_1}$, then*

$$\mathbb{P}\left[\|\boldsymbol{A}\| \leq C'''\ell\right] \geq 1 - C''\ell^{p+1} \exp\left(-c' \frac{n}{\ell}\right)$$

*for some constants $c, c', C, C'', C'''$.*

*Proof.* We partition $\boldsymbol{A}$ into blocks of size $\frac{c_2 n}{\ell}$ for an appropriately chosen $c_2$. There are $\frac{\ell}{c_2}$ such blocks, and we similarly partition the coordinates $\{1, \ldots, n\}$ into $\frac{\ell}{c_2}$ sets $K_i = \{1 + (i-1)\frac{c_2 n}{\ell} : i\frac{c_2 n}{\ell}\}$ for $i \in [\frac{\ell}{c_2}]$.

We proceed to bound the operator norm of the diagonal blocks using a standard $\varepsilon$-net argument (Vershynin, 2018). The set of unit norm vectors supported on some $K_i$ forms a sphere $\mathbb{S}^{\frac{c_2 n}{\ell}}$. We can thus construct a $\frac{1}{4}$-net $\mathcal{N}_i$ on this sphere with at most $e^{\log(9)c_2\frac{n}{\ell}}$ points. A standard argument gives

$$\|\boldsymbol{A}_{ii}\| \leq C \sup_{\boldsymbol{x} \in \mathcal{N}_i} \|\boldsymbol{x}^* \boldsymbol{A}_{ii} \boldsymbol{x}\|.$$

We control the RHS by a taking a union bound over the net, finding

$$\mathbb{P}\left[\sup_{\boldsymbol{x} \in \mathcal{N}_i} \|\boldsymbol{x}^* \boldsymbol{A}_{ii} \boldsymbol{x}\| \leq C'\right] = \mathbb{P}\left[\sup_{\boldsymbol{x} \in \mathcal{N}_i} \|\boldsymbol{x}^* \boldsymbol{A} \boldsymbol{x}\| \leq C'\right]$$

$$\geq 1 - |\mathcal{N}_i| \, C\ell^p \exp\left(-c_1 \frac{n}{\ell}\right) \geq 1 - C\ell^p \exp\left((\log(9)c_2 - c_1) \frac{n}{\ell}\right).$$

We now choose $c_2$ to satisfy $\log(9)c_2 = \frac{c_1}{2}$, and the blocks will still have non-zero size because we assume $n > \frac{2\log(9)\ell}{c_1}$. Taking a union bound over the $\frac{\ell}{c_2}$ blocks and using Lemma D.19 gives

$$\|\boldsymbol{A}\| \leq \sum_{i=1}^{\frac{\ell}{c_2}} \|\boldsymbol{A}_{ii}\| \leq C\ell$$

w.p. $\mathbb{P} \geq 1 - C'\ell^{p+1} \exp\left(-c\frac{n}{\ell}\right)$ for some constants $c, C, C'$. $\qquad\square$

**Lemma D.21.** *Assume $n \geq \max\{KL\log n, K'Ld_b, K''\}$, $d_b \geq K''' \log L$ for suitably chosen $K, K', K'', K'''$. Define $\mathcal{J}$ as in Lemma D.14. For $\boldsymbol{x} \in \mathbb{S}^{n_0 - 1}$ and*

$$\boldsymbol{\beta}_{\mathcal{J}}^{\ell} = \left(\boldsymbol{W}^{L+1} \boldsymbol{P}_{J_L} \boldsymbol{W}^L \ldots \boldsymbol{W}^{\ell+2} \boldsymbol{P}_{J_{\ell+1}}\right)^*,$$

*denote*

$$d_i = |I_i(\boldsymbol{x}) \ominus J_i|, \quad \overline{\boldsymbol{d}} = (d_1, \ldots, d_L),$$

*and $d_{\min} = \min_i d_i$. We then have*

$$\mathbb{P}\left[\mathbb{1}_{\mathcal{E}_{\delta K}} \left\|\boldsymbol{\beta}_{\mathcal{J}}^{\ell} - \boldsymbol{\beta}^{\ell}(\boldsymbol{x})\right\|_2 > C\sqrt{d_b L} + C\sqrt{s\left(\|\overline{\boldsymbol{d}}\|_1 + \frac{2\|\overline{\boldsymbol{d}}\|_1^2 Ls}{n} + \sqrt{\frac{Ld_b}{n}} \|\overline{\boldsymbol{d}}\|_1 \|\overline{\boldsymbol{d}}\|_{1/2}^{1/2}\right)}\right]$$

$$\leq e^{-c\max\{d_{\min}, 1\}s} + e^{-c'\frac{n}{L}} + e^{-c''d_b}$$

*for absolute constants $c, c', c'', C, C', C'', C'''$, where the event $\mathcal{E}_{\delta K}$ is defined in lemma D.14. Other useful forms of this result are*

$$\mathbb{P}\left[\mathbb{1}_{\mathcal{E}_{\delta K}} \left\|\boldsymbol{\beta}_{\mathcal{J}}^{\ell} - \boldsymbol{\beta}^{\ell}(\boldsymbol{x})\right\|_2 > C\sqrt{d_b L} + C\sqrt{Lns\left(L^2 s + \frac{\|\overline{\boldsymbol{d}}\|_1}{n} + \sqrt{\frac{Ld_b}{n}} \|\overline{\boldsymbol{d}}\|_{\frac{1}{2}}^{\frac{1}{2}}\right) + CL\left\langle\boldsymbol{s}, \overline{\boldsymbol{d}}\right\rangle}\right]$$

$$\leq e^{-cs} + e^{-c'\frac{n}{L}} + e^{-c''d_b} + \sum_{i=\ell}^{L} e^{-cs_i \max\{d_i, 1\}}$$

*where $s_i \geq 1$.*

*Proof.* Denoting by $\{\boldsymbol{H}^i\}$ the weight matrices projected onto the subspace orthogonal to the features as in lemma D.14, we define

$$\widehat{\boldsymbol{\beta}}_{\boldsymbol{H}}^\ell(\boldsymbol{x}) = \boldsymbol{H}^{\ell+1*}\boldsymbol{\beta}_{\boldsymbol{H}}^\ell(\boldsymbol{x}) = \left(\boldsymbol{W}^{L+1}\boldsymbol{P}_{I_L(\boldsymbol{x})}\boldsymbol{H}^L\ldots\boldsymbol{H}^{\ell+2}\boldsymbol{P}_{I_{\ell+1}(\boldsymbol{x})}\boldsymbol{H}^{\ell+1}\right)^*$$

$$\widehat{\boldsymbol{\beta}}_{\boldsymbol{H}\mathcal{J}}^\ell = \boldsymbol{H}^{\ell+1*}\boldsymbol{\beta}_{\boldsymbol{H}\mathcal{J}}^\ell = \left(\boldsymbol{W}^{L+1}\boldsymbol{P}_{J_L}\boldsymbol{H}^L\ldots\boldsymbol{H}^{\ell+2}\boldsymbol{P}_{J_{\ell+1}}\boldsymbol{H}^{\ell+1}\right)^*$$

for $\ell = 0, \ldots, L-1$. Note the additional matrix compared to the standard definition of the backward features. Control of the norm of the difference between them can then be used to control the backward features and Lipschitz constant of the network. Note also that $\boldsymbol{H}^\ell$ may not be a square matrix (and indeed in the case of the Lipschitz constant it will be rectangular). We denote the number of columns of $\boldsymbol{H}^{\ell+1}$ by $n_{\ell-1}$.

Writing

$$\mathbb{1}_{\mathcal{E}_{\delta K}}\left\|\widehat{\boldsymbol{\beta}}_{\mathcal{J}}^\ell - \widehat{\boldsymbol{\beta}}^\ell(\boldsymbol{x})\right\|_2 \leq \mathbb{1}_{\mathcal{E}_{\delta K}}\left\|\widehat{\boldsymbol{\beta}}_{\boldsymbol{H}\mathcal{J}}^\ell - \widehat{\boldsymbol{\beta}}_{\boldsymbol{H}}^\ell(\boldsymbol{x})\right\|_2 + \mathbb{1}_{\mathcal{E}_{\delta K}}\left\|\widehat{\boldsymbol{\beta}}_{\mathcal{J}}^\ell - \widehat{\boldsymbol{\beta}}_{\boldsymbol{H}\mathcal{J}}^\ell(\boldsymbol{x})\right\|_2 \\ + \mathbb{1}_{\mathcal{E}_{\delta K}}\left\|\widehat{\boldsymbol{\beta}}^\ell(\boldsymbol{x}) - \widehat{\boldsymbol{\beta}}_{\boldsymbol{H}}^\ell(\boldsymbol{x})\right\|_2, \tag{D.114}$$

we begin by bounding the first term. For $\boldsymbol{\Gamma}_{\boldsymbol{H}}^{i:j}(\boldsymbol{x}), \boldsymbol{\Gamma}_{\boldsymbol{H}\mathcal{J}}^{i:j}$ defined as in D.14 and

$$\boldsymbol{Q}^i(\boldsymbol{x}) = \boldsymbol{P}_{J_i} - \boldsymbol{P}_{I_i(\boldsymbol{x})}, \tag{D.115}$$

we have

$$\mathbb{1}_{\mathcal{E}_{\delta K}}\left\|\widehat{\boldsymbol{\beta}}_{\boldsymbol{H}\mathcal{J}}^\ell - \widehat{\boldsymbol{\beta}}_{\boldsymbol{H}}^\ell(\boldsymbol{x})\right\|_2^2 \qquad\qquad = \mathbb{1}_{\mathcal{E}_{\delta K}}\left\|\boldsymbol{W}^{L+1}\left(\boldsymbol{\Gamma}_{\boldsymbol{H}\mathcal{J}}^{L:\ell+1} - \boldsymbol{\Gamma}_{\boldsymbol{H}}^{L:\ell+1}(\boldsymbol{x})\right)\right\|_2^2$$

$$= \mathbb{1}_{\mathcal{E}_{\delta K}}\left\|\boldsymbol{W}^{L+1}\sum_{i=\ell+1}^{L}\boldsymbol{\Gamma}_{\boldsymbol{H}}^{L:i+1}\boldsymbol{Q}^i(\boldsymbol{x})\boldsymbol{H}^i\boldsymbol{\Gamma}_{\boldsymbol{H}\mathcal{J}}^{i-1:\ell+1}\right\|_2^2 \quad \doteq \left\|\sum_{i=\ell+1}^{L}\boldsymbol{b}_i\right\|_2^2.$$

We first bound $\|\boldsymbol{b}_i\|_2^2$. Repeated use of the rotational invariance of the Gaussian distribution in a similar manner to the proof of lemma D.18 gives

$$\|\boldsymbol{b}_i\|_2^2 = \mathbb{1}_{\mathcal{E}_{\delta K}}\left\|\boldsymbol{W}^{L+1}\boldsymbol{\Gamma}_{\boldsymbol{H}}^{L:i+1}\boldsymbol{Q}^i(\boldsymbol{x})\boldsymbol{H}^i\boldsymbol{\Gamma}_{\boldsymbol{H}\mathcal{J}}^{i-1:\ell+1}\right\|_2^2$$

$$\stackrel{d}{=} \frac{n}{2}\prod_{k=i+1}^{L}\underbrace{\left\|\boldsymbol{H}_{(1,:)}^{k+1}\mathbb{1}_{\mathcal{E}_{\delta K}}\boldsymbol{P}_{I_k(\boldsymbol{x})}\right\|_2^2}_{\doteq \xi_{I_k(\boldsymbol{x})}}\left\|\boldsymbol{H}_{(1,:)}^{i+1}\mathbb{1}_{\mathcal{E}_{\delta K}}\boldsymbol{Q}^i(\boldsymbol{x})\right\|_2^2\prod_{k=\ell+1}^{i-1}\underbrace{\left\|\boldsymbol{H}_{(1,:)}^{k+1}\mathbb{1}_{\mathcal{E}_{\delta K}}\boldsymbol{P}_{J_k}\right\|_2^2}_{\doteq \xi_{J_k}}\left\|\boldsymbol{H}_{(1,:)}^{\ell+1}\right\|_2^2$$

where we defined $\boldsymbol{H}_{(1,:)}^{L+1} = \sqrt{\frac{2}{n}}\boldsymbol{W}^{L+1}$. Denoting by $\tilde{\boldsymbol{W}}^k$ an independent copy of $\boldsymbol{W}^k$, rotational invariance gives

$$\xi_{J_k} \leq \left\|\tilde{\boldsymbol{W}}_{(1,:)}^{k+1}\mathbb{1}_{\mathcal{E}_{\delta K}}\boldsymbol{P}_{J_k}\right\|_2^2 \stackrel{d}{=} \frac{2}{n}\chi_k$$

where $\chi_k$ is a standard chi-squared distributed random variable with $|\text{suppdiag}\mathbb{1}_{\mathcal{E}_{\delta K}}\boldsymbol{P}_{J_k}|$ degrees of freedom that is independent of all the other variables in the problem, and similarly for $\xi_{I_k(\boldsymbol{x})}$.

A product of such terms was bounded in lemma D.18, from which we obtain

$$\mathbb{P}\left[\frac{n}{2}\prod_{k=i+1}^{L}\xi_{I_k(\boldsymbol{x})}\prod_{k=\ell+1}^{i-1}\xi_{J_k} > Cn\right] \leq e^{-c\frac{n}{L}}. \tag{D.116}$$

We similarly have

$$\left\|\boldsymbol{H}_{(1,:)}^{i+1}\mathbb{1}_{\mathcal{E}_{\delta K}}\boldsymbol{Q}^i(\boldsymbol{x})\right\|_2^2 \underset{a.s.}{\leq} \left\|\mathbb{1}_{\mathcal{E}_{\delta K}}\boldsymbol{Q}^i(\boldsymbol{x})\tilde{\boldsymbol{W}}_{(1,:)}^{k+1*}\right\|_2^2.$$

Recalling (D.115) and since

$$d_i = \left|\text{suppdiag}\boldsymbol{Q}^i(\boldsymbol{x})\right|$$

we recognize that

$$\left\|\mathbb{1}_{\mathcal{E}_{\delta K}}\boldsymbol{Q}^i(\boldsymbol{x})\tilde{\boldsymbol{W}}_{(1,:)}^{k+1*}\right\|_2^2 \stackrel{d}{=} \mathbb{1}_{\mathcal{E}_{\delta K}}\frac{2}{n}\chi_i$$

where $\chi_i$ is a standard chi-squared distributed random variable with $d_i$ degrees of freedom. If $d_i = 0$ this variable is identically 0. Otherwise, $d_i \geq 1$ and Bernstein's inequality gives

$$\mathbb{P}\left[\chi_i - d_i > Csd_i\right] \leq 2e^{-csd_i} \Rightarrow \mathbb{P}\left[\chi_i > C'sd_i\right] \leq 2e^{-csd_i} \leq 2e^{-cs\max\{d_i,1\}} \tag{D.117}$$

for some constants and $s \geq 1$. Clearly this result also holds if $d_i = 0$. Similarly, $\left\|\boldsymbol{H}_{(1,:)}^{\ell+1}\right\|_2^2$ is bounded almost surely by $\frac{2}{n}\chi_{\ell+1}$ where $\chi_{\ell+1}$ has $n_{\ell-1}$ degrees of freedom. Bernstein's inequality gives $\mathbb{P}\left[\left\|\boldsymbol{H}_{(1,:)}^{\ell+1}\right\|_2^2 > C\frac{1}{n}t\right] \leq e^{-ct}$ for $t > Kn_{\ell-1}$ for some $K$. Combining these results with (D.116) and taking a union bound, we obtain

$$\mathbb{P}\left[\sum_{i=\ell+1}^{L} \|\boldsymbol{b}_i\|_2^2 > C\frac{1}{n}t\sum_{i=\ell+1}^{L} s_i d_i\right] \leq 2\sum_{i=\ell+1}^{L} e^{-cs_i \max\{d_i,1\}} + 2L(e^{-c't} + C'e^{-c''\frac{n}{L}})$$
$$\leq 2\sum_{i=\ell+1}^{L} e^{-cs_i \max\{d_i,1\}} + e^{-c'''t} + e^{-c''''\frac{n}{L}} \tag{D.118}$$

for appropriate constants, assuming $t \geq K\log L, n \geq K'L\log L$ for some $K, K'$, which can be simplified to

$$\mathbb{P}\left[\sum_{i=\ell}^{L} \|\boldsymbol{b}_i\|_2^2 > C\frac{1}{n}ts\sum_{i=\ell+1}^{L} d_i\right] \leq 2\sum_{i=\ell}^{L} e^{-cs\max\{d_i,1\}} + e^{-c'''n_\ell t} + e^{-c''''\frac{n}{L}}$$
$$\leq 2Le^{-cs} + e^{-c'''n_\ell t} + e^{-c''''\frac{n}{L}} \leq e^{-c's} + e^{-c'''n_\ell t} + e^{-c''''\frac{n}{L}} \tag{D.119}$$

assuming $s \geq K''\log L$ for some $K''$.

We next bound $|\langle \boldsymbol{b}_i, \boldsymbol{b}_j\rangle|$ for $\ell \leq j < i \leq L$. Once again using rotational invariance starting from the last layer weights, we obtain

$$\langle \boldsymbol{b}_i, \boldsymbol{b}_j\rangle = \mathbb{1}_{\mathcal{E}_{\delta K}} \boldsymbol{W}^{L+1}\boldsymbol{\Gamma}_{\boldsymbol{H}}^{L:i+1}(\boldsymbol{x})\boldsymbol{Q}^i(\boldsymbol{x})\boldsymbol{H}^i\boldsymbol{\Gamma}_{\boldsymbol{H}\mathcal{J}}^{i-1:\ell+1}\boldsymbol{\Gamma}_{\boldsymbol{H}\mathcal{J}}^{j-1:\ell+1*}\boldsymbol{H}^{j*}\boldsymbol{Q}^j(\boldsymbol{x})\boldsymbol{\Gamma}_{\boldsymbol{H}}^{L:j+1*}(\boldsymbol{x})\boldsymbol{W}^{L+1}$$
$$\stackrel{d}{=}\frac{n}{2}\mathbb{1}_{\mathcal{E}_{\delta K}}\prod_{k=i+2}^{L}\xi_{I_k(\boldsymbol{x})}\boldsymbol{H}_{(1,:)}^{i+1}\boldsymbol{Q}^i(\boldsymbol{x})\boldsymbol{H}^i\underbrace{\boldsymbol{\Gamma}_{\boldsymbol{H}\mathcal{J}}^{i-1:\ell+1}\boldsymbol{\Gamma}_{\boldsymbol{H}\mathcal{J}}^{j-1:\ell+1*}}_{\doteq \boldsymbol{\Phi}^{i-1:j-1}}\boldsymbol{H}^{j*}\boldsymbol{Q}^j(\boldsymbol{x})\boldsymbol{\Gamma}_{\boldsymbol{H}}^{i:j+1*}(\boldsymbol{x})\boldsymbol{H}_{(:,1)}^{i+1*}$$

(where we interpret an empty product as unity). As before, we find using lemma D.18 that

$$\mathbb{P}\left[\mathbb{1}_{\mathcal{E}_{\delta K}}\frac{n}{2}\prod_{k=i+2}^{L}\xi_{I_k(\boldsymbol{x})} > Cn\right] \leq C'e^{-c\frac{n}{L}}. \tag{D.120}$$

We proceed to bound the remaining factors in $\langle \boldsymbol{b}_i, \boldsymbol{b}_j\rangle$, by first writing

$$\langle \boldsymbol{b}_i, \boldsymbol{b}_j\rangle = \mathbb{1}_{\mathcal{E}_{\delta K}}\frac{n}{2}\prod_{k=i+2}^{L}\xi_{I_k(\boldsymbol{x})}\boldsymbol{H}_{(1,:)}^{i+1}\boldsymbol{Q}^i(\boldsymbol{x})\boldsymbol{H}^i\boldsymbol{\Phi}^{i-1:j-1}\boldsymbol{H}^{j*}\boldsymbol{Q}^j(\boldsymbol{x})\boldsymbol{\Gamma}_{\boldsymbol{H}}^{i:j+1*}(\boldsymbol{x})\boldsymbol{H}_{(:,1)}^{i+1*}$$
$$\stackrel{d}{=}\mathbb{1}_{\mathcal{E}_{\delta K}}\frac{n}{2}\prod_{k=i+2}^{L}\xi_{I_k(\boldsymbol{x})}\sum_{k_i=1}^{d_i}\sum_{k_j=1}^{d_j}\boldsymbol{H}_{(1,k_i)}^{i+1}s_{k_i}\boldsymbol{H}_{(k_i,:)}^{i}\boldsymbol{\Phi}^{i-1:j-1}\boldsymbol{H}_{(:,k_j)}^{j*}s_{k_j}\boldsymbol{H}_{(k_j,1)}^{j+1*}\left\|\boldsymbol{u}^{i+1:j+1}\right\|_2$$

where $\boldsymbol{u}^{i+1:j+1} = \boldsymbol{P}_{I_{j+1}}\boldsymbol{\Gamma}_{\boldsymbol{H}}^{i:j+2*}\boldsymbol{H}_{(:,1)}^{i+1*}$ and $s_{k_m} \in \{-1,1\}$ are the signs of the elements in $\boldsymbol{Q}^m(\boldsymbol{x})$ for $m \in \{i,j\}$. In the above expression, $k_m$ index the entries on which $\mathrm{diag}\boldsymbol{Q}^m$ is supported, and we denote $d_m = |\mathrm{suppdiag}\boldsymbol{Q}^m|$ and use the permutation symmetry of the Gaussian distribution to set these to be $[d_m]$.

If $i > j + 1$, defining $\widehat{\boldsymbol{H}}^{j+1} = \widehat{\boldsymbol{W}}^{j+1}\boldsymbol{P}_{S^{j\perp}}$ where $\widehat{\boldsymbol{W}}^{j+1}$ is an independent copy of $\boldsymbol{W}^{j+1}$, with $\widehat{\boldsymbol{\Gamma}}_{\boldsymbol{H}\mathcal{J}}^{i-1:\ell+1}$ denoting the matrix $\boldsymbol{\Gamma}_{\boldsymbol{H}\mathcal{J}}^{i-1:\ell+1}$ with $\widehat{\boldsymbol{H}}_{(1,:)}^{j+1}$ in place of $\boldsymbol{H}_{(1,:)}^{j+1}$, and writing for concision

$$\Xi_{i,j}^{k_i,k_j} = \boldsymbol{H}_{(1,k_i)}^{i+1}\boldsymbol{H}_{(k_i,:)}^{i}\boldsymbol{\Gamma}_{\boldsymbol{H}\mathcal{J}}^{i-1:j+2}\boldsymbol{P}_{J_{j+1}(:,1)}\left(\boldsymbol{H}_{(1,:)}^{j+1} - \widehat{\boldsymbol{H}}_{(1,:)}^{j+1}\right)\boldsymbol{\Phi}^{j:j-1}\boldsymbol{H}_{(:,k_j)}^{j*}\boldsymbol{H}_{(k_j,1)}^{j+1*}\left\|\boldsymbol{u}^{i+1:j+1}\right\|_2$$

and

$$\Psi_{i,j}^{k_i,k_j} = H_{(1,k_i)}^{i+1} H_{(k_i,:)}^{i} \widehat{\Gamma}_{HJ}^{i-1:\ell+1} \Gamma_{HJ}^{j-1:\ell+1*} H_{(:,k_j)}^{j*} H_{(k_j,1)}^{j+1*} \left\| u^{i+1:j+1} \right\|_2$$

we have

$$\langle b_i, b_j \rangle \overset{d}{=} \begin{aligned} &\mathbb{1}_{\mathcal{E}_{\delta K}} \frac{n}{2} \prod_{k=i+2}^{L} \xi_{I_k(x)} \sum_{k_i=1}^{d_i} \sum_{k_j=1}^{d_j} \Xi_{i,j}^{k_i,j_j} \\ &+ \mathbb{1}_{\mathcal{E}_{\delta K}} \frac{n}{2} \prod_{k=i+2}^{L} \xi_{I_k(x)} \sum_{k_i=1}^{d_i} \sum_{k_j=1}^{d_j} \Psi_{i,j}^{k_i,k_j} \end{aligned} \tag{D.121}$$
$$\doteq \frac{n}{2} \left( A_1^{i,j} + A_2^{i,j} \right),$$

where we used the invariance of the Gaussian distribution to reflections around the mean, $\{H^m\} \overset{d}{=} \left\{ \tilde{W}^m P_{S^{m-1\perp}} \right\}$, and the independence between the $\left\{ \tilde{W}^m \right\}$ variables and the sign variables $\{s_{k_m}\}$ to absorb the latter into the former. Making a separate definition for concision

$$\underbrace{\mathbb{1}_{\mathcal{E}_{\delta K}} \sum_{k_i=1}^{d_i} H_{(1,k_i)}^{i+1} H_{(k_i,:)}^{i} \Gamma_{HJ}^{i-1:j+2} P_{J_{j+1}(:,1)}}_{\doteq B_1^{i,j}}$$

we first consider the term

$$A_1^{i,j} \overset{d}{=} B_1^{i,j} \left( \underbrace{\mathbb{1}_{\mathcal{E}_{\delta K}} \sum_{k_j=1}^{d_j} H_{(1,:)}^{j+1} \Phi^{j:j-1} H_{(:,k_j)}^{j*} H_{(k_j,1)}^{j+1*}}_{\doteq B_2^{i,j}} - \underbrace{\mathbb{1}_{\mathcal{E}_{\delta K}} \sum_{k_j=1}^{d_j} \widehat{H}_{(1,:)}^{j+1} \Phi^{j:j-1} H_{(:,k_j)}^{j*} H_{(k_j,1)}^{j+1*}}_{\doteq B_3^{i,j}} \right) \underbrace{\mathbb{1}_{\mathcal{E}_{\delta K}} \left\| u^{i+1:j+1} \right\|_2}_{\doteq B_4^{i,j}}.$$

Lemma D.22 gives

$$\mathbb{P}\left[ \left| B_4^{i,j} \right| > C \right] = \mathbb{P}\left[ \mathbb{1}_{\mathcal{E}_{\delta K}} \left\| u^{i+1:j+1} \right\|_2 > C \right] \leq C''' e^{-c'' \frac{n}{L}}. \tag{D.122}$$

We next consider $B_3^{i,j}$. Writing

$$B_3^{i,j} = \mathbb{1}_{\mathcal{E}_{\delta K}} \sum_{k_j=1}^{d_j} \widehat{H}_{(1,:)}^{j+1} \Phi^{j:j-1} H_{(:,k_j)}^{j*} H_{(k_j,1)}^{j+1*} \overset{d}{=} \mathbb{1}_{\mathcal{E}_{\delta K}} \sum_{k_j=1}^{d_j} \widehat{H}_{(1,:)}^{j+1} \Phi^{j:j-1} H_{(:,k_j)}^{j*} P_{S^j\perp} \tilde{W}_{(k_j,1)}^{j+1*}$$

First, since the variables $\left\{ \tilde{W}_{(k_j,1)}^{j+1*} \right\}$ are independent of $\left\{ \mathbb{1}_{\mathcal{E}_{\delta K}} \widehat{H}_{(1,:)}^{j+1} \Phi^{j:j-1} H_{(:,k_j)}^{j*} (P_{J_\ell})_{(k_j,k_j)} \right\}$, a Gaussian tail bound gives

$$\mathbb{P}\left[ \left| \sum_{k_j=1}^{d_j} \widehat{H}_{(1,:)}^{j+1} \Phi^{j:j-1} H_{(:,k_j)}^{j*} H_{(k_j,1)}^{j+1*} \right| > \sqrt{\frac{2d}{n} \sum_{k_j=1}^{d_j} \mathbb{1}_{\mathcal{E}_{\delta K}} \left( \widehat{H}_{(1,:)}^{j+1} \Phi^{j:j-1} H_{(:,k_j)}^{j*} (P_{J_\ell})_{(k_j,k_j)} \right)^2} \right]$$
$$\leq e^{-cd} \tag{D.123}$$

for some constants and $d \geq K$ for some $K$.

Two applications lemma D.22 give

$$\mathbb{P}\left[\sum_{k_j=1}^{d_j}\mathbb{1}_{\mathcal{E}_{\delta K}}\left(\widehat{\boldsymbol{H}}_{(1,:)}^{j+1}\boldsymbol{\Phi}^{j:j-1}\boldsymbol{H}_{(:,k_j)}^{j*}\left(\boldsymbol{P}_{J_\ell}\right)_{(k_j,k_j)}\right)^2 > Cd_j\frac{1}{n}t\right]$$

$$\leq \sum_{k_j=1}^{d_j}\mathbb{P}\left[\mathbb{1}_{\mathcal{E}_{\delta K}}\left(\widehat{\boldsymbol{H}}_{(1,:)}^{j+1}\boldsymbol{\Phi}^{j:j-1}\boldsymbol{H}_{(:,k_j)}^{j*}\left(\boldsymbol{P}_{J_\ell}\right)_{(k_j,k_j)}\right)^2 > C\frac{1}{n}t\right]$$

$$\leq d_j\left(e^{-c\frac{n}{L}} + e^{-c't}\right) \leq e^{-c''\frac{n}{L}} + e^{-c'''t}$$

assuming $t \geq K\log n$, $n \geq K'L\log n$ for some $K$.

Combining this bound with (D.123) we obtain

$$\mathbb{P}\left[\left|B_3^{i,j}\right| > C\frac{\sqrt{dd_jt}}{n}\right] \leq e^{-cd} + e^{-c'\frac{n}{L}} + e^{-c''t} \tag{D.124}$$

for appropriate constants.

We now turn to bounding $B_2^{i,j}$. Define by $\overline{\boldsymbol{Q}}^j$ a matrix such that $\overline{Q}_{ab}^j = \left|Q_{ab}^j(\boldsymbol{x})\right|$. Then

$$B_2^{i,j} \overset{d}{=} \mathbb{1}_{\mathcal{E}_{\delta K}}\tilde{\boldsymbol{W}}_{(1,:)}^{j+1}\boldsymbol{P}_{S^{j\perp}}\boldsymbol{\Phi}^{j:j-1}\boldsymbol{H}^{j*}\overline{\boldsymbol{Q}}^j\boldsymbol{P}_{S^{j\perp}}\tilde{\boldsymbol{W}}_{(:,1)}^{j+1*}.$$

In order to bound this term using the Hanson-Wright inequality, we first note that since

$$\left\|\boldsymbol{P}_{S^{j\perp}}\boldsymbol{\Phi}^{j:j-1}\boldsymbol{H}^{j*}\overline{\boldsymbol{Q}}^j\boldsymbol{P}_{S^{j\perp}}\right\| \leq \left\|\boldsymbol{\Phi}^{j:j-1}\boldsymbol{H}^{j*}\right\| \leq \left\|\boldsymbol{\Gamma}_{H\mathcal{J}}^{j:\ell+1}\right\|\left\|\boldsymbol{\Gamma}_{H\mathcal{J}}^{j-1:\ell+1*}\right\|\left\|\boldsymbol{H}^{j*}\right\|,$$

$$\left\|\boldsymbol{P}_{S^{j\perp}}\boldsymbol{\Phi}^{j:j-1}\boldsymbol{H}^{j*}\overline{\boldsymbol{Q}}^j\boldsymbol{P}_{S^{j\perp}}\right\|_F^2 \leq \left\|\boldsymbol{\Phi}^{j:j-1}\boldsymbol{H}^{j*}\right\|^2\left\|\overline{\boldsymbol{Q}}^j\right\|_F^2 = \left\|\boldsymbol{\Phi}^{j:j-1}\boldsymbol{H}^{j*}\right\|^2 d_j,$$

we can use lemma D.14, a standard $\varepsilon$-net argument to control the operator norm of a Gaussian matrix and a union bound to obtain

$$\mathbb{P}\left[\begin{matrix}\left\{\left\|\mathbb{1}_{\mathcal{E}_{\delta K}}\boldsymbol{P}_{S^{j\perp}}\boldsymbol{\Phi}^{j:j-1}\boldsymbol{H}^{j*}\overline{\boldsymbol{Q}}^j\boldsymbol{P}_{S^{j\perp}}\right\| \leq CL\right\}\\ \cap\left\{\left\|\mathbb{1}_{\mathcal{E}_{\delta K}}\boldsymbol{P}_{S^{j\perp}}\boldsymbol{\Phi}^{j:j-1}\boldsymbol{H}^{j*}\overline{\boldsymbol{Q}}^j\boldsymbol{P}_{S^{j\perp}}\right\|_F^2 \leq CL^2d_j\right\}\end{matrix}\right] \geq 1 - e^{-c\frac{n}{L}} + e^{-c'n} \geq 1 - e^{-c''\frac{n}{L}}.$$

We also have

$$\underset{\tilde{\boldsymbol{W}}^{j+1}}{\mathbb{E}}\tilde{\boldsymbol{W}}_{(1,:)}^{j+1}\boldsymbol{P}_{S^{j\perp}}\boldsymbol{\Phi}^{j:j-1}\boldsymbol{H}^{j*}\overline{\boldsymbol{Q}}^j\boldsymbol{P}_{S^{j\perp}}\tilde{\boldsymbol{W}}_{(:,1)}^{j+1*} = \frac{2}{n}\text{tr}\left[\boldsymbol{P}_{S^{j\perp}}\boldsymbol{\Phi}^{j:j-1}\boldsymbol{H}^{j*}\overline{\boldsymbol{Q}}^j\boldsymbol{P}_{S^{j\perp}}\right]$$

$$= \frac{2}{n}\sum_{k_j=1}^{d_j}\widehat{\boldsymbol{e}}_{k_j}^*\boldsymbol{P}_{S^{j\perp}}\boldsymbol{\Gamma}_{H\mathcal{J}}^{j:\ell+1}\boldsymbol{\Gamma}_{H\mathcal{J}}^{j-1:\ell+1*}\boldsymbol{H}^{j*}\widehat{\boldsymbol{e}}_{k_j}$$

and using lemmas D.14 and D.22 gives

$$\mathbb{P}\left[\left|\sum_{k_j=1}^{d_j}\widehat{\boldsymbol{e}}_{k_j}^*\boldsymbol{P}_{S^{j\perp}}\boldsymbol{\Gamma}_{H\mathcal{J}}^{j:\ell+1}\boldsymbol{\Gamma}_{H\mathcal{J}}^{j-1:\ell+1*}\boldsymbol{H}^{j*}\widehat{\boldsymbol{e}}_{k_j}\right| \leq C\frac{d_j}{n}\right] \geq 1 - d_je^{-c\frac{n}{L}} \geq 1 - e^{-c'\frac{n}{L}}.$$

assuming $n > KL\log n$ for some $K$. Denoting the union of these two events by $\mathcal{G}$, an application of the Hanson-Wright inequality (lemma (G.4)) gives

$$\mathbb{P}\left[\mathbb{1}_{\mathcal{G}}\left|B_2^{i,j}\right| > s_2 + \frac{2Cd_j}{n}\right] \leq C\exp\left(-c\min\left\{\frac{n^2s_2^2}{L^2d_j}, \frac{ns_2}{L}\right\}\right) \tag{D.125}$$

for appropriate constants and $s_2 \geq 0$, and an additional union bound gives

$$\mathbb{P}\left[\left|B_2^{i,j}\right| > s_2 + \frac{2Cd_j}{n}\right] \leq C\exp\left(-c\min\left\{\frac{n^2s_2^2}{L^2d_j}, \frac{ns_2}{L}\right\}\right) + e^{-\frac{n}{L}}. \tag{D.126}$$

We next turn to bounding $\left|B_1^{i,j}\right|$. Rotational invariance of the Gaussian distribution gives

$$B_1^{i,j} = \mathbb{1}_{\mathcal{E}_{\delta K}} \sum_{k_i=1}^{d_i} \boldsymbol{H}_{(1,k_i)}^{i+1} \boldsymbol{H}_{(k_i,:)}^{i} \boldsymbol{\Gamma}_{\boldsymbol{HJ}}^{i-1:j+2} \boldsymbol{P}_{J_{j+1}(:,1)}$$

$$\stackrel{d}{=} \mathbb{1}_{\mathcal{E}_{\delta K}} \sum_{k_i=1}^{d_i} \boldsymbol{H}_{(1,k_i)}^{i+1} \tilde{\boldsymbol{W}}_{(k_i,1)}^{i} \left\| \boldsymbol{P}_{J_{i-1}} \boldsymbol{\Gamma}_{\boldsymbol{HJ}}^{i-1:j+2} \boldsymbol{P}_{J_{j+1}(:,1)} \right\|_2$$

since $\boldsymbol{P}_{J_{i-1}} \boldsymbol{\Gamma}_{\boldsymbol{HJ}}^{i-1:j+2} \boldsymbol{P}_{J_{j+1}(:,1)}$ and $\tilde{\boldsymbol{W}}_{(k_i,1)}^{i}$ are independent.

Since $\boldsymbol{H}_{(1,k_i)}^{i+1}$, $\tilde{\boldsymbol{W}}_{(1,k_i)}^{i}$ are both sub-Gaussian with sub-Gaussian norm bounded by $\frac{C'}{\sqrt{n}}$, the product of two such variables is a sub-exponential variable with sub exponential norm satisfying $\left\| \boldsymbol{H}_{(1,k_i)}^{i+1} \tilde{\boldsymbol{W}}_{(k_i,1)}^{i} \right\|_{\psi_1} \leq \frac{C}{n}$ for some constants. Thus the first sum above is a sum of independent, zero-mean sub-exponential random variables, and Bernstein's inequality gives

$$\mathbb{P}\left[ \left| \sum_{k_i=1}^{d_i} \boldsymbol{H}_{(1,k_i)}^{i+1} \tilde{\boldsymbol{W}}_{(k_i,1)}^{i} \right| > s_1 \right] \leq 2e^{-c\min\{\frac{n^2 s_1^2}{d_i}, ns_1\}} \tag{D.127}$$

for $s_1 \geq 1$ and some constant $c$.

Since $\left\| \mathbb{1}_{\mathcal{E}_{\delta K}} \boldsymbol{P}_{J_{i-1}} \boldsymbol{\Gamma}_{\boldsymbol{HJ}}^{i-1:j+2} \boldsymbol{P}_{J_{j+1}(:,1)} \right\|_2 \leq \left\| \mathbb{1}_{\mathcal{E}_{\delta K}} \boldsymbol{\Gamma}_{\boldsymbol{HJ}}^{i-1:j+2} \widehat{\boldsymbol{e}}_1 \right\|_2$ we can apply lemma D.14 to obtain

$$\mathbb{P}\left[ \left\| \mathbb{1}_{\mathcal{E}_{\delta K}} \boldsymbol{P}_{J_{i-1}} \boldsymbol{\Gamma}_{\boldsymbol{HJ}}^{i-1:j+2} \boldsymbol{P}_{J_{j+1}(:,1)} \right\|_2 > C \right] \leq C' e^{-c\frac{n}{L}}$$

for appropriate constants. Combining the last two results gives

$$\mathbb{P}\left[ \left| B_1^{i,j} \right| > Cs_1 \right] \leq e^{-c\min\{\frac{n^2 s_1^2}{d_i}, ns_1\}} + e^{-c'\frac{n}{L}} \tag{D.128}$$

for some constants.

Combining the above with (D.122), (D.124) and (D.126), we have

$$\mathbb{P}\left[ \left| A_1^{ij} \right| \geq C \left( s_2 + \frac{2d_j}{n} + \frac{\sqrt{dd_j}t}{n} \right) s_1 \right]$$
$$\leq e^{-c\min\{\frac{n^2 s_1^2}{d_i}, ns_1\}} + e^{-c'\frac{n}{L}} + e^{-c''d} + e^{-c'''\min\{\frac{n^2 s_2^2}{L^2 d_j}, \frac{ns_2}{L}\}} + e^{-c''''t} \tag{D.129}$$

In the above proof we assumed $i > j + 1$. If instead $i = j + 1$ we simply set $\widehat{\boldsymbol{\Gamma}}_{\boldsymbol{HJ}}^{i-1:\ell+1} = \boldsymbol{\Gamma}_{\boldsymbol{HJ}}^{i-1:\ell+1}$ in the expression for $A_2^{i,j}$ in (D.121) and we have $\langle \boldsymbol{b}_{j+1}, \boldsymbol{b}_j \rangle = A_2^{j+1,j}$.

We now turn to controlling the term $A_2^{i,j}$. Since $i > j$, $\boldsymbol{H}_{(k_i,:)}^{i} \stackrel{d}{=} \tilde{\boldsymbol{W}}_{(k_i,:)}^{i} \boldsymbol{P}_{S^{i-1}\perp}$ and $\boldsymbol{P}_{S^{i-1}\perp} \widehat{\boldsymbol{\Gamma}}_{\boldsymbol{HJ}}^{i-1:\ell+1} \boldsymbol{\Gamma}_{\boldsymbol{HJ}}^{j-1:\ell+1*} \boldsymbol{H}_{(:,k_j)}^{j*}$ is independent of $\tilde{\boldsymbol{W}}_{(k_i,:)}^{i}$, rotational invariance of the Gaussian distribution gives

$$A_2^{i,j} \stackrel{d}{=} \underbrace{\sum_{k_i=1}^{d_i} \boldsymbol{H}_{(1,k_i)}^{i+1} \tilde{\boldsymbol{W}}_{(k_i,1)}^{i}}_{\doteq C_1^i}$$
$$\times \underbrace{\mathbb{1}_{\mathcal{E}_{\delta K}} \left( \sum_{k_j=1}^{d_j} \left\| \boldsymbol{P}_{S^{i-1}\perp} \widehat{\boldsymbol{\Gamma}}_{\boldsymbol{HJ}}^{i-1:\ell+1} \boldsymbol{\Gamma}_{\boldsymbol{HJ}}^{j-1:\ell+1*} \boldsymbol{H}_{(:,k_j)}^{j*} \right\|_2 \boldsymbol{H}_{(k_j,1)}^{j+1*} \right) \left\| \boldsymbol{u}^{i+1:j+1} \right\|_2}_{\doteq C_2^{i,j}}. \tag{D.130}$$

We bound $\left| C_1^i \right|$ using (D.127).

It remains to control $\left|C_2^{i,j}\right|$. Since $\boldsymbol{H}_{(k_j,1)}^{j+1*} \overset{d}{=} (\boldsymbol{P}_{S^{j\perp}})_{(k_j,k_j)} \tilde{\boldsymbol{W}}_{(1,k_j)}^{j+1}$ and $\tilde{\boldsymbol{W}}_{(1,k_j)}^{j+1}$ are independent of $(\boldsymbol{P}_{S^{j\perp}})_{(k_j,k_j)} \left\| \boldsymbol{P}_{S^{i-1\perp}} \widehat{\boldsymbol{\Gamma}}_{\boldsymbol{HJ}}^{i-1:\ell+1} \boldsymbol{\Gamma}_{\boldsymbol{HJ}}^{j-1:\ell+1*} \boldsymbol{H}_{(:,k_j)}^{j*} \right\|_2$, the second factor in (D.130) is also zero-mean, and it follows that

$$
\mathbb{P}\left[ \begin{array}{c} \left| \mathbb{1}_{\mathcal{E}_{\delta K}} \sum_{k_j=1}^{d_j} (\boldsymbol{P}_{S^{j\perp}})_{(k_j,k_j)} \left\| \boldsymbol{P}_{S^{i-1\perp}} \widehat{\boldsymbol{\Gamma}}_{\boldsymbol{HJ}}^{i-1:\ell+1} \boldsymbol{\Gamma}_{\boldsymbol{HJ}}^{j-1:\ell+1*} \boldsymbol{H}_{(:,k_j)}^{j*} \right\|_2 \tilde{\boldsymbol{W}}_{(1,k_j)}^{j+1} \right| \\ > \sqrt{\frac{2d}{n} \sum_{k_j=1}^{d_j} \mathbb{1}_{\mathcal{E}_{\delta K}} (\boldsymbol{P}_{S^{j\perp}})_{(k_j,k_j)} \left\| \boldsymbol{P}_{S^{i-1\perp}} \widehat{\boldsymbol{\Gamma}}_{\boldsymbol{HJ}}^{i-1:\ell+1} \boldsymbol{\Gamma}_{\boldsymbol{HJ}}^{j-1:\ell+1*} \boldsymbol{H}_{(:,k_j)}^{j*} \right\|_2^2} \end{array} \right] \le C' e^{-cd}
$$

for some constants and $d \ge 0$. Since $\left\| \boldsymbol{P}_{S^{i-1\perp}} \widehat{\boldsymbol{\Gamma}}_{\boldsymbol{HJ}}^{i-1:\ell+1} \boldsymbol{\Gamma}_{\boldsymbol{HJ}}^{j-1:\ell+1*} \boldsymbol{H}_{(:,k_j)}^{j*} \right\|_2^2 \le \left\| \widehat{\boldsymbol{\Gamma}}_{\boldsymbol{HJ}}^{i-1:\ell+1} \right\|^2 \left\| \boldsymbol{\Gamma}_{\boldsymbol{HJ}}^{j-1:\ell+1*} \boldsymbol{H}_{(:,k_j)}^{j*} \right\|_2^2$, applying lemmas D.22 and D.14 total of $d_j$ times and taking a union bound gives

$$
\mathbb{P}\left[ \sum_{k_j=1}^{d_j} \mathbb{1}_{\mathcal{E}_{\delta K}} (\boldsymbol{P}_{S^{j\perp}})_{(k_j,k_j)} \left\| \boldsymbol{P}_{S^{i-1\perp}} \widehat{\boldsymbol{\Gamma}}_{\boldsymbol{HJ}}^{i-1:\ell+1} \boldsymbol{\Gamma}_{\boldsymbol{HJ}}^{j-1:\ell+1*} \boldsymbol{H}_{(:,k_j)}^{j*} \right\|_2^2 > C \frac{d_j L t}{n} \right]
$$
$$
\le d_j \left( e^{-c\frac{n}{L}} + e^{-c't} \right) \le e^{-c'\frac{n}{L}} + e^{-c't}
$$

where we assumed $n \ge KL \log n, t \ge K' \log n$ for some constants. Combining the above three results with (D.122) and taking a union bound, we obtain

$$
\mathbb{P}\left[ \left| A_2^{i,j} \right| > C s_1 \frac{\sqrt{d_j d n_{\ell-1} L t}}{n} \right] \le e^{-cd} + e^{-c'\frac{n}{L}} + e^{-c''t} + e^{-c'''t} + e^{-c''''\min\{\frac{n^2 s_1^2}{d_i}, n s_1\}}
$$

for appropriate constants.

Taking a union bound over this result as well as (D.129) and (D.120) allows us to bound the inner product by

$$
\mathbb{P}\left[ |\langle \boldsymbol{b}_i, \boldsymbol{b}_j \rangle| \ge C n s_1 \left( s_2 + \frac{d_j}{n} + \frac{\sqrt{d_j d n_{\ell-1} L t}}{n} \right) \right]
$$
$$
\le e^{-c\min\{\frac{n^2 s_1^2}{d_i}, n s_1\}} + e^{-c'\frac{n}{L}} + e^{-cd} + e^{-c'''\min\left\{\frac{n^2 s_2^2}{L^2 d_j}, \frac{n s_2}{L}\right\}} + e^{-c''''t} \tag{D.131}
$$

for some constants, again assuming $n \ge KLd$. At this point we obtain a bound on the sum of these inner products that will be useful in an application where the $\{d_i\}$ are expected to be small. Subsequently, we will derive a different expression that will be useful when they are large.

We now choose $s_1 = \frac{d_i s}{n}, s_2 = \frac{d_j L s}{n}, t = \frac{n}{n_{\ell-1}}$ for some $s \ge 1$, which gives

$$
\mathbb{P}\left[ |\langle \boldsymbol{b}_i, \boldsymbol{b}_j \rangle| \ge C d_i s \left( \frac{d_j L s}{n} + \frac{d_j}{n} + \sqrt{\frac{L d d_j}{n}} \right) \right] \le C' e^{-c\min\{d_i, d_i\}s} + C'' e^{-c'\frac{n}{L}} + C''' e^{-c''d}
$$

for appropriately chosen constants. Note that if $d_i = 0$ or $d_j = 0$ then $|\langle \boldsymbol{b}_i, \boldsymbol{b}_j \rangle|$ is identically zero, hence we can replace the term $\min\{d_i, d_i\}$ in the tail above by $\max\{1, \min\{d_i, d_i\}\}$ and the result will still hold if $d_i = 0$ or $d_j = 0$. Lower bounding the second expression by $d_{\min} = \min_i d_i$ gives

$$
\mathbb{P}\left[ |\langle \boldsymbol{b}_i, \boldsymbol{b}_j \rangle| \ge C d_i s \left( \frac{2 d_j L s}{n} + \sqrt{\frac{L d d_j}{n}} \right) \right] \le C' e^{-c\max\{d_{\min}, 1\}s} + C'' e^{-c'\frac{n}{L}} + C''' e^{-c''d}.
$$

Recalling the definition of $\overline{\boldsymbol{d}}$ in the lemma statement, an additional union bound over the values of $i, j$ in the expression above combined with (D.119) gives

$$
\mathbb{P}\left[\mathbb{1}_{\mathcal{E}_{\delta K}}\left\|\widehat{\boldsymbol{\beta}}_{\boldsymbol{H}\mathcal{J}}^{\ell} - \widehat{\boldsymbol{\beta}}_{\boldsymbol{H}}^{\ell}(\boldsymbol{x})\right\|_2^2 > Cs\left(\|\overline{\boldsymbol{d}}\|_1 + \frac{2\|\overline{\boldsymbol{d}}\|_1^2 Ls}{n} + \sqrt{\frac{Ld}{n}}\|\overline{\boldsymbol{d}}\|_1\|\overline{\boldsymbol{d}}\|_{1/2}^{1/2}\right)\right]
$$

$$
\leq \mathbb{P}\left[\mathbb{1}_{\mathcal{E}_{\delta K}}\left\|\widehat{\boldsymbol{\beta}}_{\boldsymbol{H}\mathcal{J}}^{\ell} - \widehat{\boldsymbol{\beta}}_{\boldsymbol{H}}^{\ell}(\boldsymbol{x})\right\|_2^2 > Cs\sum_{i,j=\ell+1, i\neq j}^{L} d_i\left(\frac{2d_j Ls}{n} + \sqrt{\frac{Ldd_j}{n}}\right) + Cs\sum_{i=\ell+1}^{L} d_i\right]
$$

$$
\leq L^2\left(C'e^{-c\max\{d_{\min},1\}s} + C''e^{-c'\frac{n}{L}} + C'''e^{-c''d}\right)
$$

$$
\leq C'e^{-c'''\max\{d_{\min},1\}s} + C''e^{-c''''\frac{n}{L}} + C'''e^{-c'''''d}
$$

$$
C'e^{-c'''s} + C''e^{-c''''\frac{n}{L}} + C'''e^{-c'''''d}
$$

$$(\text{D.132})$$

for appropriate constants, where we assumed $d \geq K\log L$, $s \geq \max\{1, K'\log L\}$, $n \geq K''L\log L$ for some $K, K', K''$. Taking a square root gives a bound on the first term in (D.114).

We next consider a different bound for this term that will be useful when the $\{d_i\}$ are large. Our starting point will be D.131. If we set $s_1 = s, s_2 = Ls$ and use (D.118) we obtain

$$
\mathbb{P}\left[\mathbb{1}_{\mathcal{E}_{\delta K}}\left\|\widehat{\boldsymbol{\beta}}_{\boldsymbol{H}\mathcal{J}}^{\ell} - \widehat{\boldsymbol{\beta}}_{\boldsymbol{H}}^{\ell}(\boldsymbol{x})\right\|_2^2 > CLns\left(L^2 s + \frac{\|\overline{\boldsymbol{d}}\|_1}{n} + \frac{\sqrt{Ldt}}{n}\|\overline{\boldsymbol{d}}\|_{1/2}^{1/2}\right) + C\frac{t}{n}\sum_{i=\ell+1}^{L} s_i d_i\right]
$$

$$
\leq \mathbb{P}\left[\mathbb{1}_{\mathcal{E}_{\delta K}}\left\|\widehat{\boldsymbol{\beta}}_{\boldsymbol{H}\mathcal{J}}^{\ell} - \widehat{\boldsymbol{\beta}}_{\boldsymbol{H}}^{\ell}(\boldsymbol{x})\right\|_2^2 > Cns\sum_{i,j=\ell+1, i\neq j}^{L}\left(Ls + \frac{d_j}{n} + \frac{\sqrt{Ldd_j t}}{n}\right) + C\frac{t}{n}\sum_{i=\ell+1}^{L} s_i d_i\right]
$$

$$
\leq L^2\left(C'\exp\left(-\tilde{c}\min\{\frac{n^2 s^2}{\|\overline{\boldsymbol{d}}\|_\infty}, ns\}\right) + C''e^{-\tilde{c}'\frac{n}{L}} + C'''e^{-\tilde{c}''d}\right)
$$

$$
+ \sum_{i=\ell+1}^{L} e^{-\tilde{c}s_i\max\{d_i,1\}} + e^{-\tilde{c}'''t}
$$

$$
\leq e^{-cns} + e^{-c'\frac{n}{L}} + e^{-c''d} + \sum_{i=\ell+1}^{L} e^{-c'''s_i\max\{d_i,1\}} + e^{-c''''t}
$$

$$(\text{D.133})$$

for appropriate constants under similar assumptions on $n, L, d$.

To bound the remaining terms in (D.114), since

$$
\mathbb{1}_{\mathcal{E}_{\delta K}}\left\|\widehat{\boldsymbol{\beta}}_{\mathcal{J}}^{\ell} - \widehat{\boldsymbol{\beta}}_{\boldsymbol{H}\mathcal{J}}^{\ell}\right\|_2 \leq \mathbb{1}_{\mathcal{E}_{\delta K}}\left\|\boldsymbol{W}^{L+1}\left(\boldsymbol{\Gamma}_{\mathcal{J}}^{L:\ell+1} - \boldsymbol{\Gamma}_{\boldsymbol{H}\mathcal{J}}^{L:\ell+1}\right)\right\|_2\left\|\boldsymbol{H}^{\ell+1}\right\|
$$

$$
\overset{d}{=} \mathbb{1}_{\mathcal{E}_{\delta K}}\left\|\left(\boldsymbol{\Gamma}_{\mathcal{J}}^{L:\ell+1} - \boldsymbol{\Gamma}_{\boldsymbol{H}\mathcal{J}}^{L:\ell+1}\right)\boldsymbol{W}^{L+1*}\right\|_2\left\|\boldsymbol{H}^{\ell+1}\right\|
$$

and we can apply lemma D.14 and an $\varepsilon$-net argument to bound the first and second factors respectively, to conclude

$$
\mathbb{P}\left[\mathbb{1}_{\mathcal{E}_{\delta K}}\left\|\widehat{\boldsymbol{\beta}}_{\mathcal{J}}^{\ell} - \widehat{\boldsymbol{\beta}}_{\boldsymbol{H}\mathcal{J}}^{\ell}\right\|_2 > C\sqrt{dL}\right] \leq C'e^{-cd} + Ce^{c'n} \leq e^{-cd}
$$

for some $d$ such that $d \geq K\log L$ and assuming $n > K'd$. An identical result holds for the last term in (D.114) where we simply choose $J_i = I_i(\boldsymbol{x})$ for all $i$. In conclusion, using (D.132) we have

$$
\mathbb{P}\left[\mathbb{1}_{\mathcal{E}_{\delta K}}\left\|\widehat{\boldsymbol{\beta}}_{\mathcal{J}}^{\ell} - \widehat{\boldsymbol{\beta}}^{\ell}(\boldsymbol{x})\right\|_2 > C\sqrt{dL} + C\sqrt{s\left(\|\overline{\boldsymbol{d}}\|_1 + \frac{2\|\overline{\boldsymbol{d}}\|_1^2 Ls}{n} + \sqrt{\frac{Ld}{n}}\|\overline{\boldsymbol{d}}\|_1\|\overline{\boldsymbol{d}}\|_{1/2}^{1/2}\right)}\right]
$$

$$
\leq C'e^{-cs} + C''e^{-c'\frac{n}{L}} + C'''e^{-c''d}
$$

$$(\text{D.134})$$

for appropriate constants, while if we use (D.133) instead we obtain

$$\mathbb{P}\left[\mathbb{1}_{\mathcal{E}_{\delta K}}\left\|\widehat{\boldsymbol{\beta}}_{\mathcal{J}}^{\ell}-\widehat{\boldsymbol{\beta}}^{\ell}(\boldsymbol{x})\right\|_{2}>C\sqrt{dL}+C\sqrt{Lns\left(L^{2}s+\frac{\|\overline{\boldsymbol{d}}\|_{1}}{n}+\frac{\sqrt{Ldt}}{n}\|\overline{\boldsymbol{d}}\|_{\frac{1}{2}}^{\frac{1}{2}}\right)+\frac{CLt}{n}\left\langle\boldsymbol{s},\overline{\boldsymbol{d}}\right\rangle}\right]$$
$$\leq e^{-cs}+e^{-c'\frac{n}{L}}+e^{-c''d}+\sum_{i=\ell+1}^{L}e^{-c'''s_{i}\max\{d_{i},1\}}+e^{-c''''t}$$

(D.135)

It remains to transfer control from $\left\|\widehat{\boldsymbol{\beta}}_{H\mathcal{J}}^{\ell}-\widehat{\boldsymbol{\beta}}_{H}^{\ell}(\boldsymbol{x})\right\|_{2}$ to $\left\|\boldsymbol{\beta}_{H\mathcal{J}}^{\ell}-\boldsymbol{\beta}_{H}^{\ell}(\boldsymbol{x})\right\|_{2}$. Note that

$$\left\|\widehat{\boldsymbol{\beta}}_{H\mathcal{J}}^{\ell}-\widehat{\boldsymbol{\beta}}_{H}^{\ell}(\boldsymbol{x})\right\|_{2}^{2}=\left\|\boldsymbol{H}^{\ell+1*}\left(\boldsymbol{\beta}_{H\mathcal{J}}^{\ell}-\boldsymbol{\beta}_{H}^{\ell}(\boldsymbol{x})\right)\right\|_{2}^{2}\overset{d}{=}\left\|\boldsymbol{P}_{S^{\ell\perp}}\tilde{\boldsymbol{W}}_{(:,1)}^{\ell+1*}\right\|_{2}^{2}\left\|\boldsymbol{\beta}_{H\mathcal{J}}^{\ell}-\boldsymbol{\beta}_{H}^{\ell}(\boldsymbol{x})\right\|_{2}^{2}$$

where if $\ell=0$ we define $\boldsymbol{P}_{S^{0\perp}}=\boldsymbol{I}_{n_{0}\times n_{0}}$. Since $\underset{\tilde{\boldsymbol{W}}^{\ell+1}}{\mathbb{E}}\left\|\boldsymbol{P}_{S^{\ell\perp}}\tilde{\boldsymbol{W}}_{(:,1)}^{\ell+1}\right\|_{2}^{2}=\frac{2}{n}\operatorname{tr}\left[\boldsymbol{P}_{S^{\ell\perp}}\right]=\frac{2(n-1)}{n}$, Bernstein's inequality gives (assuming $n>K$ for some $K$)

$$\mathbb{P}\left[\left\|\boldsymbol{P}_{S^{\ell\perp}}\tilde{\boldsymbol{W}}_{(:,1)}^{\ell+1*}\right\|_{2}^{2}<\frac{1}{C}\right]\leq e^{-cn},$$

and hence with the same probability

$$\left\|\boldsymbol{\beta}_{H\mathcal{J}}^{\ell}-\boldsymbol{\beta}_{H}^{\ell}(\boldsymbol{x})\right\|_{2}^{2}\leq\frac{\left\|\widehat{\boldsymbol{\beta}}_{H\mathcal{J}}^{\ell}-\widehat{\boldsymbol{\beta}}_{H}^{\ell}(\boldsymbol{x})\right\|_{2}^{2}}{\left\|\boldsymbol{P}_{S^{\ell\perp}}\tilde{\boldsymbol{W}}_{(:,1)}^{\ell+1*}\right\|_{2}^{2}}\leq C\left\|\widehat{\boldsymbol{\beta}}_{H\mathcal{J}}^{\ell}-\widehat{\boldsymbol{\beta}}_{H}^{\ell}(\boldsymbol{x})\right\|_{2}^{2}.$$

The bounds (D.134), (D.135) also apply to $\left\|\boldsymbol{\beta}_{H\mathcal{J}}^{\ell}-\boldsymbol{\beta}_{H}^{\ell}(\boldsymbol{x})\right\|_{2}$ up to a constant factor, with the same probability up to a $e^{-cn}$ term which we can absorb into the existing tail by demanding $n\geq KL$ for some $K$. $\qquad\square$

**Lemma D.22.** *For any $\ell+1<m\leq j+1$, $k\in[n]$, if $n\geq KL$ for some $K$ then*

$$\mathbb{P}\left[\mathbb{1}_{\mathcal{E}_{\delta K}}\left\|\boldsymbol{H}_{(k,:)}^{j+1}\boldsymbol{\Gamma}_{H\mathcal{J}}^{j:m}\right\|_{2}>C\right]\leq e^{-c\frac{n}{L}}$$

*and if $m=\ell+1$*

$$\mathbb{P}\left[\mathbb{1}_{\mathcal{E}_{\delta K}}\left\|\boldsymbol{H}_{(k,:)}^{j+1}\boldsymbol{\Gamma}_{H\mathcal{J}}^{j:\ell+1}\right\|_{2}>C\sqrt{\frac{1}{n}t}\right]\leq e^{-c\frac{n}{L}}+e^{-c't}$$

*assuming $t\geq K'n_{\ell-1}$ for some $K'$.*

*Proof.* If $j+1>m$,

$$\mathbb{1}_{\mathcal{E}_{\delta K}}\left\|\boldsymbol{H}_{(1,:)}^{j+1}\boldsymbol{\Gamma}_{H\mathcal{J}}^{j:m}\right\|_{2}=\mathbb{1}_{\mathcal{E}_{\delta K}}\left\|\tilde{\boldsymbol{W}}_{(1,:)}^{j+1}\boldsymbol{P}_{S^{j\perp}}\boldsymbol{\Gamma}_{H\mathcal{J}}^{j:m+1}\boldsymbol{P}_{J_{m}}\boldsymbol{H}^{m}\right\|_{2}$$

$$\overset{d}{=}\mathbb{1}_{\mathcal{E}_{\delta K}}\left\|\tilde{\boldsymbol{W}}_{(1,:)}^{j+1}\boldsymbol{P}_{S^{j\perp}}\boldsymbol{\Gamma}_{H\mathcal{J}}^{j:m+1}\boldsymbol{P}_{J_{m}}\tilde{\boldsymbol{W}}^{m}\boldsymbol{P}_{S^{m-1\perp}}\right\|_{2}\quad\leq\mathbb{1}_{\mathcal{E}_{\delta K}}\left\|\tilde{\boldsymbol{W}}_{(1,:)}^{j+1}\boldsymbol{P}_{S^{j\perp}}\boldsymbol{\Gamma}_{H\mathcal{J}}^{j:m+1}\boldsymbol{P}_{J_{m}}\tilde{\boldsymbol{W}}^{m}\right\|_{2}$$

$$\overset{d}{=}\mathbb{1}_{\mathcal{E}_{\delta K}}\left\|\tilde{\boldsymbol{W}}_{(1,:)}^{j+1}\boldsymbol{P}_{S^{j\perp}}\boldsymbol{\Gamma}_{H\mathcal{J}}^{j:m+1}\boldsymbol{P}_{J_{m}}\right\|_{2}\left\|\tilde{\boldsymbol{W}}_{(1,:)}^{m}\right\|_{2}\quad\leq\mathbb{1}_{\mathcal{E}_{\delta K}}\left\|\boldsymbol{P}_{S^{j\perp}}\boldsymbol{\Gamma}_{H\mathcal{J}}^{j:m+1}\tilde{\boldsymbol{W}}_{(1,:)}^{j+1*}\right\|_{2}\left\|\tilde{\boldsymbol{W}}_{(1,:)}^{m}\right\|_{2}$$

$$\overset{d}{=}\left\|\boldsymbol{P}_{S^{j\perp}}\boldsymbol{\Gamma}_{H\mathcal{J}}^{j:m+1}\widehat{\boldsymbol{e}}_{1}\right\|_{2}\left\|\tilde{\boldsymbol{W}}_{(1,:)}^{j+1*}\right\|_{2}\left\|\tilde{\boldsymbol{W}}_{(1,:)}^{m}\right\|_{2}.$$

If on the other hand $j+1=m$, we have $\mathbb{1}_{\mathcal{E}_{\delta K}}\left\|\boldsymbol{H}_{(1,:)}^{j+1}\boldsymbol{\Gamma}_{H\mathcal{J}}^{j:m}\right\|_{2}=\mathbb{1}_{\mathcal{E}_{\delta K}}\left\|\boldsymbol{H}_{(1,:)}^{j+1}\right\|_{2}\leq\left\|\tilde{\boldsymbol{W}}_{(1,:)}^{j+1}\right\|_{2}$. Bernstein's inequality gives $\mathbb{P}\left[\left\|\tilde{\boldsymbol{W}}_{(1,:)}^{j+1*}\right\|_{2}>C\right]\leq C'e^{-cn}$ and $\mathbb{P}\left[\left\|\tilde{\boldsymbol{W}}_{(1,:)}^{\ell+1}\right\|_{2}>C\sqrt{\frac{1}{n}t}\right]\leq 2e^{-ct}$ for $t\geq 1$, while lemma D.14 gives

$$\mathbb{P}\left[\left\|\mathbb{1}_{\mathcal{E}_{\delta K}}\boldsymbol{\Gamma}_{H\mathcal{J}}^{j:m+1}\widehat{\boldsymbol{e}}_{1}\right\|_{2}>C\right]\leq C'e^{-c\frac{n}{L}}.$$

Taking union bounds gives the desired results. $\qquad\square$

**Lemma D.23** (Generalized backward features inner product concentration). *Fix $\boldsymbol{x}, \boldsymbol{x}' \in \mathbb{S}^{n_0-1}$, $\nu = \angle(\boldsymbol{x}, \boldsymbol{x}')$. Define a collection of support sets $\mathcal{J}$, generalized backward features $\boldsymbol{\beta}_{\mathcal{J}}^{\ell}$, a constant $\delta_s$ and event $\mathcal{E}_{\delta K}$ as in lemma D.14. Assuming $n \geq \max\{KL\log n, K'\}$, $d \geq K''\log n$ for suitably chosen $K, K', K''$, we have*

$$
\mathbb{P}\left[\exists \ell \in [L] : \begin{array}{l} \left| \mathbb{1}_{\mathcal{E}_{\delta K}} \left\langle \boldsymbol{\beta}_{\mathcal{J}}^{\ell}, \boldsymbol{\beta}_{\mathcal{J}'}^{\ell} \right\rangle - \frac{n}{2} \prod_{\ell'=\ell}^{L-1} \left(1 - \frac{\varphi^{(\ell')}(\nu)}{\pi}\right) \right| \\ > C\left(d^2\sqrt{Ln} + \sqrt{d\delta_s Ln + d^{3/2}\delta_s\left(1 + \frac{\delta_s}{\sqrt{n}}\right)L^{5/2}}\right) \end{array}\right] \leq C'e^{-cd}
$$

*for absolute constants $c, C, C'$. If additionally we have $\mathbb{P}[\mathcal{E}_{\delta K}] \geq 1 - e^{-c'd}$ then the same result holds without the truncation on $\mathbb{1}_{\mathcal{E}_{\delta K}}$, with worse constants.*

*Proof.* Note that

$$
\left| \mathbb{1}_{\mathcal{E}_{\delta K}} \left\langle \boldsymbol{\beta}_{\mathcal{J}}^{\ell}, \boldsymbol{\beta}_{\mathcal{J}'}^{\ell} \right\rangle - \frac{n}{2} \prod_{\ell'=\ell}^{L-1} \left(1 - \frac{\varphi^{(\ell')}(\nu)}{\pi}\right) \right|
$$
$$
\leq \left| \mathbb{1}_{\mathcal{E}_{\delta K}} \left\langle \boldsymbol{\beta}_{\mathcal{J}}^{\ell} - \boldsymbol{\beta}^{\ell}(\boldsymbol{x}), \boldsymbol{\beta}_{\mathcal{J}'}^{\ell} \right\rangle \right| + \left| \mathbb{1}_{\mathcal{E}_{\delta K}} \left\langle \boldsymbol{\beta}^{\ell}(\boldsymbol{x}), \boldsymbol{\beta}_{\mathcal{J}'}^{\ell} - \boldsymbol{\beta}^{\ell}(\boldsymbol{x}') \right\rangle \right|
$$
$$
\qquad + \left| \mathbb{1}_{\mathcal{E}_{\delta K}} \left\langle \boldsymbol{\beta}^{\ell}(\boldsymbol{x}), \boldsymbol{\beta}^{\ell}(\boldsymbol{x}') \right\rangle - \frac{n}{2} \prod_{\ell'=\ell}^{L-1} \left(1 - \frac{\varphi^{(\ell')}(\nu)}{\pi}\right) \right| \tag{D.136}
$$
$$
\leq \begin{array}{l} \mathbb{1}_{\mathcal{E}_{\delta K}} \left\| \boldsymbol{\beta}_{\mathcal{J}'}^{\ell}(\boldsymbol{x}') \right\|_2 \left\| \boldsymbol{\beta}_{\mathcal{J}}^{\ell} - \boldsymbol{\beta}^{\ell}(\boldsymbol{x}) \right\|_2 + \mathbb{1}_{\mathcal{E}_{\delta K}} \left\| \boldsymbol{\beta}^{\ell}(\boldsymbol{x}) \right\|_2 \left\| \boldsymbol{\beta}_{\mathcal{J}'}^{\ell} - \boldsymbol{\beta}^{\ell}(\boldsymbol{x}') \right\|_2 \\ + \left| \mathbb{1}_{\mathcal{E}_{\delta K}} \left\langle \boldsymbol{\beta}^{\ell}(\boldsymbol{x}), \boldsymbol{\beta}^{\ell}(\boldsymbol{x}') \right\rangle - \frac{n}{2} \prod_{\ell'=\ell}^{L-1} \left(1 - \frac{\varphi^{(\ell')}(\nu)}{\pi}\right) \right| \end{array}.
$$

In order to bound the first two terms, we use rotational invariance of the Gaussian distribution twice to obtain

$$
\mathbb{1}_{\mathcal{E}_{\delta K}} \left\| \boldsymbol{\beta}_{\mathcal{J}'}^{\ell} \right\|_2^2 = \mathbb{1}_{\mathcal{E}_{\delta K}} \left\| \boldsymbol{W}^{L+1} \boldsymbol{\Gamma}_{\mathcal{J}(\boldsymbol{x}')}^{L:\ell+1} \boldsymbol{P}_{J'_{\ell}(\boldsymbol{x}')} \right\|_2^2
$$
$$
\underset{a.s.}{\leq} \mathbb{1}_{\mathcal{E}_{\delta K}} \left\| \boldsymbol{W}^{L+1} \boldsymbol{\Gamma}_{\mathcal{J}'}^{L:\ell+1} \right\|_2^2 \overset{d}{=} \mathbb{1}_{\mathcal{E}_{\delta K}} \left\| \boldsymbol{\Gamma}_{\mathcal{J}'}^{L:\ell+1} \boldsymbol{W}^{L+1} \right\|_2^2 \overset{d}{=} \left\| \boldsymbol{W}^{L+1} \right\|_2^2 \mathbb{1}_{\mathcal{E}_{\delta K}} \left\| \boldsymbol{\Gamma}_{\mathcal{J}'}^{L:\ell+1} \widehat{\boldsymbol{e}}_1 \right\|_2^2.
$$

Bernstein's inequality gives $\mathbb{P}\left[\left\| \boldsymbol{W}^{L+1} \right\|_2^2 > Cn\right] \leq 2e^{-cn}$, while $\mathbb{1}_{\mathcal{E}_{\delta K}} \left\| \boldsymbol{\Gamma}_{\mathcal{J}'}^{L:\ell+1} \widehat{\boldsymbol{e}}_1 \right\|_2^2$ can be bounded using D.14 to give

$$
\mathbb{P}\left[\mathbb{1}_{\mathcal{E}_{\delta K}} \left\| \boldsymbol{\beta}_{\mathcal{J}'}^{\ell} \right\|_2^2 > Cn\right] \leq C'e^{-cn} + C''e^{-c'\frac{n}{L}} \leq C'''e^{-c''\frac{n}{L}}
$$

for appropriate constants. Using lemma D.21 to bound $\mathbb{1}_{\mathcal{E}_{\delta K}} \left\| \boldsymbol{\beta}_{\mathcal{J}}^{\ell} - \boldsymbol{\beta}^{\ell}(\boldsymbol{x}) \right\|_2$ we obtain

$$
\mathbb{P}\left[\mathbb{1}_{\mathcal{E}_{\delta K}} \left\| \boldsymbol{\beta}_{\mathcal{J}'}^{\ell}(\boldsymbol{x}') \right\|_2 \left\| \boldsymbol{\beta}_{\mathcal{J}}^{\ell} - \boldsymbol{\beta}^{\ell}(\boldsymbol{x}) \right\|_2 > C\sqrt{dLn} + C'\sqrt{d\delta_s Ln + d^{3/2}\delta_s\left(1 + \frac{\delta_s}{\sqrt{n}}\right)L^{5/2}}\right]
$$
$$
\leq C''e^{-c\frac{n}{L}} + C'''e^{-c'd} \leq C''''e^{-c''d}
$$

for some constants, assuming $n \geq KLd$ for some $K$. Bounding the second term in (D.136) in an identical fashion and the last term in (D.136) using Lemma D.4 we obtain

$$
\mathbb{P}\left[\begin{array}{l} \left| \mathbb{1}_{\mathcal{E}_{\delta K}} \left\langle \boldsymbol{\beta}_{\mathcal{J}}^{\ell}, \boldsymbol{\beta}_{\mathcal{J}'}^{\ell} \right\rangle - \frac{n}{2} \prod_{\ell'=\ell}^{L-1} \left(1 - \frac{\varphi^{(\ell')}(\nu)}{\pi}\right) \right| \\ > C\left(d^2\sqrt{Ln} + \sqrt{d\delta_s Ln + d^{3/2}\delta_s\left(1 + \frac{\delta_s}{\sqrt{n}}\right)L^{5/2}}\right) \end{array}\right]
$$
$$
\leq \mathbb{P}\left[\begin{array}{l} \left| \mathbb{1}_{\mathcal{E}_{\delta K}} \left\langle \boldsymbol{\beta}_{\mathcal{J}}^{\ell}, \boldsymbol{\beta}_{\mathcal{J}'}^{\ell} \right\rangle - \frac{n}{2} \prod_{\ell'=\ell}^{L-1} \left(1 - \frac{\varphi^{(\ell')}(\nu)}{\pi}\right) \right| \\ > C'\left(\sqrt{dLn} + \sqrt{d\delta_s Ln + d^{3/2}\delta_s\left(1 + \frac{\delta_s}{\sqrt{n}}\right)L^{5/2}}\right) + C'd^2\sqrt{Ln} \end{array}\right]
$$

$$\leq \quad \mathbb{P}\left[\begin{array}{c} \left|\mathbb{1}_{\mathcal{E}_{\delta K}}\left\|\boldsymbol{\beta}_{\mathcal{J}'}^{\ell}\right\|_2 \left\|\boldsymbol{\beta}_{\mathcal{J}}^{\ell}-\boldsymbol{\beta}^{\ell}(\boldsymbol{x})\right\|_2 + \mathbb{1}_{\mathcal{E}_{\delta K}}\left\|\boldsymbol{\beta}_{\mathcal{J}}^{\ell}\right\|_2 \left\|\boldsymbol{\beta}_{\mathcal{J}'}^{\ell}-\boldsymbol{\beta}^{\ell}(\boldsymbol{x}')\right\|_2\right| \\ > C'\left(\sqrt{dLn}+\sqrt{d\delta_s Ln+d^{3/2}\delta_s\left(1+\frac{\delta_s}{\sqrt{n}}\right)L^{5/2}}\right) \end{array}\right]$$

$$+\mathbb{P}\left[\left|\mathbb{1}_{\mathcal{E}_{\delta K}}\left\langle\boldsymbol{\beta}_{\mathcal{J}}^{\ell},\boldsymbol{\beta}_{\mathcal{J}'}^{\ell}\right\rangle - \frac{n}{2}\prod_{\ell'=\ell}^{L-1}\left(1-\frac{\varphi^{(\ell')}(\nu)}{\pi}\right)\right| > C'd^2\sqrt{Ln}\right]$$

$$\leq C''e^{-cd}+C'''e^{-c'd}\leq C''''e^{-c''d}.$$

for appropriate constants assuming $d \geq 1$. Taking a union bound over all possible choices of $\ell \in [L]$ and using $d \geq K\log L$ for some $K$ gives the desired result. If we additionally have $\mathbb{P}\left[\mathcal{E}_{\delta K}\right] \geq 1 - e^{-c'''d}$ for some $c'''$, we can write

$$\left|\left\langle\boldsymbol{\beta}_{\mathcal{J}}^{\ell},\boldsymbol{\beta}_{\mathcal{J}'}^{\ell}\right\rangle - \frac{n}{2}\prod_{\ell'=\ell}^{L-1}\left(1-\frac{\varphi^{(\ell')}(\nu)}{\pi}\right)\right| = \left|\mathbb{1}_{\mathcal{E}_{\delta K}}\left\langle\boldsymbol{\beta}_{\mathcal{J}}^{\ell},\boldsymbol{\beta}_{\mathcal{J}'}^{\ell}\right\rangle - \frac{n}{2}\prod_{\ell'=\ell}^{L-1}\left(1-\frac{\varphi^{(\ell')}(\nu)}{\pi}\right)\right|$$

$$+ \left|(1-\mathbb{1}_{\mathcal{E}_{\delta K}})\left\langle\boldsymbol{\beta}_{\mathcal{J}}^{\ell},\boldsymbol{\beta}_{\mathcal{J}'}^{\ell}\right\rangle\right|$$

and since the last term is zero w.p. $\geq 1 - e^{-c'''d}$ we obtain the same result as in the truncated case, with possibly worse constants. $\qquad\square$

## D.4 AUXILIARY RESULTS

**Lemma D.24.** *There are absolute constants $c_1, C, C' > 0$ and absolute constants $K, K' > 0$ such that for any $L \in \mathbb{N}$, if $n \geq \max\{K\log^4 n, K'L\}$, then for every $\ell \in [L]$ one has*

$$\left|\mathbb{E}\left[\varphi^{(L-\ell)}(\hat{\nu}^\ell) - \varphi^{(L-\ell+1)}(\hat{\nu}^{\ell-1}) \,\middle|\, \mathcal{F}^{\ell-1}\right]\right| \leq C\frac{\log n}{n}\frac{\hat{\nu}^{\ell-1}}{1+(c_0/64)(L-\ell)\hat{\nu}^{\ell-1}}\left(1+\log L\right)+\frac{C'}{n^2}.$$

*The constant $c_1$ is the absolute constant appearing in Lemma E.1.*

*Proof.* The case of $\ell = L$ follows immediately from Lemma E.1 with an appropriate choice of $d \geq K''$ for $K'' > 0$ some absolute constant. Henceforth we assume $\ell \in [L-1]$. We Taylor expand (with Lagrange remainder) the smooth function $\varphi^{(L-\ell)}$ about the point $\varphi(\hat{\nu}^{\ell-1})$, obtaining for any $t \in [0,\pi]$

$$\varphi^{(L-\ell)}(t) = \varphi^{(L-\ell+1)}(\hat{\nu}^{\ell-1}) + \dot{\varphi}^{(L-\ell)}(\varphi(\hat{\nu}^{\ell-1}))\left(t - \varphi(\hat{\nu}^{\ell-1})\right) + \frac{\ddot{\varphi}^{(L-\ell)}(\xi)}{2}\left(t - \varphi(\hat{\nu}^{\ell-1})\right)^2,$$

where $\xi$ is some point of $[0,\pi]$ lying in between $t$ and $\varphi(\hat{\nu}^{\ell-1})$. In particular, putting $t = \hat{\nu}^\ell$, we obtain

$$\varphi^{(L-\ell)}(\hat{\nu}^\ell)-\varphi^{(L-\ell+1)}(\hat{\nu}^{\ell-1}) = \dot{\varphi}^{(L-\ell)}(\varphi(\hat{\nu}^{\ell-1}))\left(\hat{\nu}^\ell - \varphi(\hat{\nu}^{\ell-1})\right)+\frac{\ddot{\varphi}^{(L-\ell)}(\xi)}{2}\left(\hat{\nu}^\ell - \varphi(\hat{\nu}^{\ell-1})\right)^2,$$

where $\xi$ is some point of $[0,\pi]$ lying in between $\hat{\nu}^\ell$ and $\varphi(\hat{\nu}^{\ell-1})$. By (C.23) and (C.26), we have that $\ddot{\varphi}^{(L-\ell)} \leq 0$, whence

$$\varphi^{(L-\ell)}(\hat{\nu}^\ell) - \varphi^{(L-\ell+1)}(\hat{\nu}^{\ell-1}) \leq \dot{\varphi}^{(L-\ell)}(\varphi(\hat{\nu}^{\ell-1}))\left(\hat{\nu}^\ell - \varphi(\hat{\nu}^{\ell-1})\right). \tag{D.137}$$

Using Lemma E.5 and an induction, we have that $\dot{\varphi}^{(L-\ell)}$ is decreasing, and moreover by the concavity property we have $\varphi(\hat{\nu}^{\ell-1}) \geq \hat{\nu}^{\ell-1}/2$. An application of Lemmas E.1 and C.15 then yields

$$\mathbb{E}\left[\varphi^{(L-\ell)}(\hat{\nu}^\ell) - \varphi^{(L-\ell+1)}(\hat{\nu}^{\ell-1}) \,\middle|\, \mathcal{F}^{\ell-1}\right] \leq \left(C\hat{\nu}^{\ell-1}\frac{\log n}{n}+C'n^{-c_1 d}\right)\frac{1}{1+(c_0/4)(L-\ell)\hat{\nu}^{\ell-1}}$$

$$\leq C\frac{\log n}{n}\frac{\hat{\nu}^{\ell-1}}{1+(c_0/4)(L-\ell)\hat{\nu}^{\ell-1}}+C'n^{-c_1 d},$$

as long as $d \geq K$ and $n \geq K'd^4\log^4 n$. In particular, we can choose $d = \max\{K, 2/c_1\}$ to obtained the claimed error for the upper bound. Next, for the lower bound, we make use of the estimate

$$\ddot{\varphi}^{(L-\ell)}(\nu) \geq \underbrace{-\frac{C}{1+(c_0/8)(L-\ell)\nu}\left(1+\frac{1}{(c_0/8)\nu}\log\left(1+(c_0/8)(L-\ell-1)\nu\right)\right)}_{f(\nu)},$$

which follows from Lemma C.16 and $\ddot{\varphi}^{(L-\ell)} \leq 0$; by that lemma, we have that $f$ is increasing. By Lemma E.3, as long as $n \geq K' \log^4 n$, there is an event $\mathcal{E}$ on which $|\hat{\nu}^\ell - \varphi(\hat{\nu}^{\ell-1})| \leq C\hat{\nu}^{\ell-1}\sqrt{\log n / n} + C'n^{-3}$ and which satisfies $\mathbb{P}[\mathcal{E} \mid \mathcal{F}^{\ell-1}] \geq 1 - C''n^{-3}$. In particular, on the event $\mathcal{E}$ we have $\hat{\nu}^\ell \geq \hat{\nu}^{\ell-1}/4 - C'/n^3$ provided $n \geq 16C^2 \log n$, and so on the event $\mathcal{E}$ we have $\xi \geq \min\{\varphi(\hat{\nu}^{\ell-1}), \hat{\nu}^{\ell-1}/4 - C'/n^3\} \geq \hat{\nu}^{\ell-1}/4 - C'/n^3$. We can thus write

$$\varphi^{(L-\ell)}(\hat{\nu}^\ell) - \varphi^{(L-\ell+1)}(\hat{\nu}^{\ell-1})$$

$$\geq \dot{\varphi}^{(L-\ell)}(\varphi(\hat{\nu}^{\ell-1}))\left(\hat{\nu}^\ell - \varphi(\hat{\nu}^{\ell-1})\right) + \frac{f(\xi)}{2}\left(\hat{\nu}^\ell - \varphi(\hat{\nu}^{\ell-1})\right)^2$$

$$= \dot{\varphi}^{(L-\ell)}(\varphi(\hat{\nu}^{\ell-1}))\left(\hat{\nu}^\ell - \varphi(\hat{\nu}^{\ell-1})\right) + (\mathbb{1}_{\mathcal{E}} + \mathbb{1}_{\mathcal{E}^c})\frac{f(\xi)}{2}\left(\hat{\nu}^\ell - \varphi(\hat{\nu}^{\ell-1})\right)^2$$

$$\geq \dot{\varphi}^{(L-\ell)}(\varphi(\hat{\nu}^{\ell-1}))\left(\hat{\nu}^\ell - \varphi(\hat{\nu}^{\ell-1})\right) + \mathbb{1}_{\mathcal{E}}\frac{f(\frac{\hat{\nu}^{\ell-1}}{4} - \frac{C_4}{n^3})}{2}\left(\hat{\nu}^\ell - \varphi(\hat{\nu}^{\ell-1})\right)^2 - (2C'''\pi^2 L)\mathbb{1}_{\mathcal{E}^c}$$

$$\geq \dot{\varphi}^{(L-\ell)}(\varphi(\hat{\nu}^{\ell-1}))\left(\hat{\nu}^\ell - \varphi(\hat{\nu}^{\ell-1})\right) + \frac{f(\frac{\hat{\nu}^{\ell-1}}{4} - \frac{C_4}{n^3})}{2}\left(\hat{\nu}^\ell - \varphi(\hat{\nu}^{\ell-1})\right)^2 - (2C'''\pi^2 L)\mathbb{1}_{\mathcal{E}^c}$$

where the inequality in the third line follows from boundedness of the angles and the magnitude estimate on $f$ in Lemma C.16, together with our estimate on $\xi$ on $\mathcal{E}$, and the inequality in the final line is a consequence of $f \leq 0$, which allows us to drop the indicator for $\mathcal{E}$ and obtain a lower bound. Taking conditional expectations using the previous lower bound and applying $\mathcal{F}^{\ell-1}$-measurability of $\hat{\nu}^{\ell-1}$ and boundedness of the angles together with our conditional measure bound on $\mathcal{E}^c$, we obtain

$$\mathbb{E}\left[\varphi^{(L-\ell)}(\hat{\nu}^\ell) - \varphi^{(L-\ell+1)}(\hat{\nu}^{\ell-1}) \,\Big|\, \mathcal{F}^{\ell-1}\right] \geq -C\frac{\log n}{n}\frac{\hat{\nu}^{\ell-1}}{1 + (c_0/4)(L-\ell)\hat{\nu}^{\ell-1}} - \frac{C'}{n^2} - \frac{C_5 L}{n^3}$$
$$+ \frac{f(\frac{\hat{\nu}^{\ell-1}}{4} - \frac{C_4}{n^3})}{2}\mathbb{E}\left[\left(\hat{\nu}^\ell - \varphi(\hat{\nu}^{\ell-1})\right)^2 \,\Big|\, \mathcal{F}^{\ell-1}\right],$$

where we also apply the complementary bound obtained by our previous work following (D.137). Since the $CL$ estimate in Lemma C.16 applies also to $\dot{f}$, and since $f \leq 0$, an application of Lemma E.4 with an appropriate choice of $d$ and the choice $n \geq K' \log^4 n$ then yields (with a larger absolute constant $C'$)

$$\mathbb{E}\left[\varphi^{(L-\ell)}(\hat{\nu}^\ell) - \varphi^{(L-\ell+1)}(\hat{\nu}^{\ell-1}) \,\Big|\, \mathcal{F}^{\ell-1}\right] \geq -C\frac{\log n}{n}\frac{\hat{\nu}^{\ell-1}}{1 + (c_0/4)(L-\ell)\hat{\nu}^{\ell-1}} - \frac{C'}{n^2} - \frac{C_6 L}{n^3}$$
$$+ \frac{C_7 \log n}{n}\left(\hat{\nu}^{\ell-1}\right)^2 f\left(\frac{\hat{\nu}^{\ell-1}}{4} - \frac{C_4}{n^3}\right).$$

If we choose $n \geq (C_6/C')L$, we can simplify this last estimate to

$$\mathbb{E}\left[\varphi^{(L-\ell)}(\hat{\nu}^\ell) - \varphi^{(L-\ell+1)}(\hat{\nu}^{\ell-1}) \,\Big|\, \mathcal{F}^{\ell-1}\right] \geq -C\frac{\log n}{n}\frac{\hat{\nu}^{\ell-1}}{1 + (c_0/4)(L-\ell)\hat{\nu}^{\ell-1}} - \frac{2C'}{n^2}$$
$$+ \frac{C_7 \log n}{n}\left(\hat{\nu}^{\ell-1}\right)^2 f\left(\frac{\hat{\nu}^{\ell-1}}{4} - \frac{C_4}{n^3}\right).$$

To conclude, we divide our analysis into two cases: when $\hat{\nu}^{\ell-1} \geq 8C_4/n^3$, we have $\hat{\nu}^{\ell-1}/4 - C_4 n^{-3} \geq \hat{\nu}^{\ell-1}/8$, and so

$$\left(\hat{\nu}^{\ell-1}\right)^2 f\left(\frac{\hat{\nu}^{\ell-1}}{4} - \frac{C_4}{n^3}\right) \geq \left(\hat{\nu}^{\ell-1}\right)^2 f\left(\frac{\hat{\nu}^{\ell-1}}{8}\right)$$

$$= 64\left(\frac{\hat{\nu}^{\ell-1}}{8}\right)^2 f\left(\frac{\hat{\nu}^{\ell-1}}{8}\right)$$

$$\geq -\frac{8C\pi\hat{\nu}^{\ell-1}}{1 + (c_0/64)(L-\ell)\hat{\nu}^{\ell-1}}\left(1 + \frac{8\log(L-\ell)}{c_0\pi}\right),$$

where the last inequality follows from Lemma C.16. On the other hand, when $\hat{\nu}^{\ell-1} \leq 8C_4/n^3$, the $CL$ estimate in Lemma C.16 implies

$$\left(\hat{\nu}^{\ell-1}\right)^2 f\left(\frac{\hat{\nu}^{\ell-1}}{4} - \frac{C_4}{n^3}\right) \geq -\frac{64CC_4^2 L}{n^6} \geq -\frac{64CC_4^2 L}{n^3} \geq -\frac{2C'}{n^2},$$

where the last estimate holds when $n \geq (32CC_4^2/C')L$. Adding these two estimates together, we obtain one that is valid regardless of the value of $\hat{\nu}^{\ell-1}$, and choosing $n \geq C_7 \log n$ to combine the residuals, we obtain (after worst-case adjusting the constants)

$$\mathbb{E}\Big[\varphi^{(L-\ell)}(\hat{\nu}^\ell) - \varphi^{(L-\ell+1)}(\hat{\nu}^{\ell-1}) \,\Big|\, \mathcal{F}^{\ell-1}\Big] \geq -\frac{4C'}{n^2} - \frac{C_8 \log n}{n} \frac{\hat{\nu}^{\ell-1}\,(1 + \log(L - \ell))}{1 + (c_0/64)(L - \ell)\hat{\nu}^{\ell-1}}.$$

Combining with our previous work, we obtain

$$\left|\mathbb{E}\Big[\varphi^{(L-\ell)}(\hat{\nu}^\ell) - \varphi^{(L-\ell+1)}(\hat{\nu}^{\ell-1}) \,\Big|\, \mathcal{F}^{\ell-1}\Big]\right| \leq C\frac{\log n}{n}\frac{\hat{\nu}^{\ell-1}\,(1 + \log L)}{1 + (c_0/64)(L - \ell)\hat{\nu}^{\ell-1}} + C'\frac{1}{n^2}$$

after worst-casing constants. $\qquad\square$

**Lemma D.25.** *There are absolute constants $c_1, C, C', C'', C''' > 0$ and absolute constants $K, K' > 0$ such that for any $d \geq K$, if $n \geq K'd^4 \log^4 n$, then for every $L \in \mathbb{N}$ and every $\ell \in [L]$ one has*

$$\mathbb{P}\left[\left|\varphi^{(L-\ell)}(\hat{\nu}^\ell) - \varphi^{(L-\ell+1)}(\hat{\nu}^{\ell-1})\right| \leq \sqrt{\frac{d \log n}{n}} \frac{2C\hat{\nu}^{\ell-1}}{1 + (c_0/8)(L - \ell)\hat{\nu}^{\ell-1}} + 2C'n^{-c_1 d/2} \,\middle|\, \mathcal{F}^{\ell-1}\right]$$
$$\geq 1 - C'''n^{-c_1 d},$$

*and*

$$\mathbb{E}\left[\left(\varphi^{(L-\ell)}(\hat{\nu}^\ell) - \varphi^{(L-\ell+1)}(\hat{\nu}^{\ell-1})\right)^2 \,\middle|\, \mathcal{F}^{\ell-1}\right] \leq 4C^2 \frac{d \log n}{n}\left(\frac{\hat{\nu}^{\ell-1}}{1 + (c_0/8)(L - \ell)\hat{\nu}^{\ell-1}}\right)^2$$
$$+ C''n^{-c_1 d/2}.$$

*The constant $c_1$ is the absolute constant appearing in Lemma E.1.*

*Proof.* We will fix the meaning of the absolute constants $C, C', C'' > 0$ throughout the proof below. By Lemma E.3, we have if $d \geq K$ and $n \geq K'd^4 \log^4 n$ that for every $\ell \in [L]$

$$\mathbb{P}\left[\left|\hat{\nu}^\ell - \varphi(\hat{\nu}^{\ell-1})\right| \leq C\hat{\nu}^{\ell-1}\sqrt{\frac{d \log n}{n}} + C'n^{-c_1 d} \,\middle|\, \mathcal{F}^{\ell-1}\right] \geq 1 - C''n^{-c_1 d}. \qquad (D.138)$$

By Lemma C.15, we have the estimate

$$\left|\dot{\varphi}^{(\ell)}(t)\right| \leq \frac{1}{1 + (c_0/2)\ell t},$$

valid for any $\ell \in \mathbb{N}_0$. Writing $\Xi^\ell = \hat{\nu}^\ell - \varphi(\hat{\nu}^{\ell-1})$ so that $\hat{\nu}^\ell = \varphi(\hat{\nu}^{\ell-1}) + \Xi^\ell$, we have that $(\Xi^\ell)$ is adapted to $(\mathcal{F}^\ell)$, and by the fundamental theorem of calculus

$$\left|\varphi^{(L-\ell)}(\hat{\nu}^\ell) - \varphi^{(L-\ell+1)}(\hat{\nu}^{\ell-1})\right| = \left|\int_{\varphi(\hat{\nu}^{\ell-1})+\Xi^\ell}^{\varphi(\hat{\nu}^{\ell-1})} \frac{dt}{1 + (c_0/2)(L - \ell)t}\right|.$$

The integrand is nonnegative, so by Jensen's inequality we have

$$\left(\varphi^{(L-\ell)}(\hat{\nu}^\ell) - \varphi^{(L-\ell+1)}(\hat{\nu}^{\ell-1})\right)^2 \leq |\Xi^\ell| \int_{\varphi(\hat{\nu}^{\ell-1})+\Xi^\ell}^{\varphi(\hat{\nu}^{\ell-1})} \frac{dt}{(1 + (c_0/2)(L - \ell)t)^2},$$

and an integration then yields

$$\left(\varphi^{(L-\ell)}(\hat{\nu}^\ell) - \varphi^{(L-\ell+1)}(\hat{\nu}^{\ell-1})\right)^2$$
$$\leq \frac{\left(\Xi^\ell\right)^2}{|1 + (c_0/2)(L - \ell)\varphi(\hat{\nu}^{\ell-1})||1 + (c_0/2)(L - \ell)(\varphi(\hat{\nu}^{\ell-1}) + \Xi^\ell)|}. \qquad (D.139)$$

Choosing $d \geq 1/c_1$, we can guarantee that whenever $\nu \geq \frac{C'}{C}n^{-c_1 d/2}$, one has

$$C\nu\sqrt{\frac{d \log n}{n}} + C'n^{-c_1 d} \leq 2C\nu\sqrt{\frac{d \log n}{n}}, \qquad (D.140)$$

and choosing $n \geq 64C^2 d \log n$, we can guarantee that

$$\frac{\nu}{2} - 2C\nu\sqrt{\frac{d \log n}{n}} \geq \frac{\nu}{4}. \tag{D.141}$$

In particular, the last condition guarantees $2C\sqrt{d \log n / n} \leq 1/4$. By concavity of $\varphi$ via Lemma E.5, we have $\varphi(\hat{\nu}^{\ell-1}) \geq \hat{\nu}^{\ell-1}/2$, and using (D.138) to obtain

$$\mathbb{P}\left[ |\Xi^\ell| \leq C\hat{\nu}^{\ell-1}\sqrt{\frac{d \log n}{n}} + C'n^{-c_1 d} \,\middle|\, \mathcal{F}^{\ell-1} \right] \geq 1 - C''n^{-c_1 d},$$

we have by (D.140) and (D.141) as well as the concavity lower bound on $\varphi$

$$\mathbb{P}\left[ \varphi(\hat{\nu}^{\ell-1}) + \Xi^\ell \geq \hat{\nu}^{\ell-1}/4 \,\middle|\, \mathcal{F}^{\ell-1} \right] \geq 1 - C''n^{-c_1 d}$$

as long as $\hat{\nu}^{\ell-1} \geq (C'/C)n^{-c_1 d/2}$. In particular, plugging these bounds into (D.139) and taking square roots, we obtain by a union bound

$$\mathbb{P}\left[ \left| \varphi^{(L-\ell)}(\hat{\nu}^\ell) - \varphi^{(L-\ell+1)}(\hat{\nu}^{\ell-1}) \right| \leq 2C\sqrt{\frac{d \log n}{n}}\left| \frac{\hat{\nu}^{\ell-1}}{1 + (c_0/8)(L-\ell)\hat{\nu}^{\ell-1}} \right| \,\middle|\, \mathcal{F}^{\ell-1} \right]$$

$$\geq 1 - 2C''n^{-cd}$$

whenever $\hat{\nu}^{\ell-1} \geq (C'/C)n^{-c_1 d/2}$. Meanwhile, when $\hat{\nu}^{\ell-1} \leq (C'/C)n^{-c_1 d/2}$, if we choose $n \geq d \log n$ we have

$$C\hat{\nu}^{\ell-1}\sqrt{\frac{d \log n}{n}} + C'n^{-c_1 d} \leq 2C'n^{-c_1 d/2},$$

and we can use the 1-Lipschitz property of $\varphi^{(L-\ell)}$, which follows from Lemma E.5, to obtain using (D.138)

$$\mathbb{P}\left[ \left| \varphi^{(L-\ell)}(\hat{\nu}^\ell) - \varphi^{(L-\ell+1)}(\hat{\nu}^{\ell-1}) \right| \leq 2C'n^{-c_1 d/2} \,\middle|\, \mathcal{F}^{\ell-1} \right]$$

$$\geq \mathbb{P}\left[ \left| \varphi^{(L-\ell)}(\hat{\nu}^\ell) - \varphi^{(L-\ell+1)}(\hat{\nu}^{\ell-1}) \right| \leq C\hat{\nu}^{\ell-1}\sqrt{\frac{d \log n}{n}} + C'n^{-c_1 d} \,\middle|\, \mathcal{F}^{\ell-1} \right]$$

$$\geq \mathbb{P}\left[ \left| \hat{\nu}^\ell - \varphi(\hat{\nu}^{\ell-1}) \right| \leq C\hat{\nu}^{\ell-1}\sqrt{\frac{d \log n}{n}} + C'n^{-c_1 d} \,\middle|\, \mathcal{F}^{\ell-1} \right]$$

$$\geq 1 - C''n^{-c_1 d}.$$

Because $|\varphi^{(L-\ell)}(\hat{\nu}^\ell) - \varphi^{(L-\ell+1)}(\hat{\nu}^{\ell-1})| \geq 0$, we can then obtain using a union bound

$$\mathbb{P}\left[ \left| \varphi^{(L-\ell)}(\hat{\nu}^\ell) - \varphi^{(L-\ell+1)}(\hat{\nu}^{\ell-1}) \right| \leq \sqrt{\frac{d \log n}{n}}\left| \frac{2C\hat{\nu}^{\ell-1}}{1 + (c_0/8)(L-\ell)\hat{\nu}^{\ell-1}} \right| + \frac{2C'}{n^{c_1 d/2}} \,\middle|\, \mathcal{F}^{\ell-1} \right]$$

$$\geq 1 - 3C''n^{-c_1 d},$$

which holds regardless of the value of $\hat{\nu}^{\ell-1}$. We can then obtain the second bound using this one, via a partition of the expectation: let

$$\mathcal{E} = \left\{ \left| \varphi^{(L-\ell)}(\hat{\nu}^\ell) - \varphi^{(L-\ell+1)}(\hat{\nu}^{\ell-1}) \right| \leq \sqrt{\frac{d \log n}{n}}\left| \frac{2C\hat{\nu}^{\ell-1}}{1 + (c_0/8)(L-\ell)\hat{\nu}^{\ell-1}} \right| + 2C'n^{-c_1 d/2} \right\},$$

so that $\mathcal{E} \in \mathcal{F}^\ell$, and $\mathbb{P}[\mathcal{E} \mid \mathcal{F}^{\ell-1}] \geq 1 - 3C''n^{-c_1 d}$ by our work above. Then we have

$$\mathbb{E}\left[ \left( \varphi^{(L-\ell)}(\hat{\nu}^\ell) - \varphi^{(L-\ell+1)}(\hat{\nu}^{\ell-1}) \right)^2 \,\middle|\, \mathcal{F}^{\ell-1} \right]$$

$$\leq \mathbb{E}\left[ \left( \varphi^{(L-\ell)}(\hat{\nu}^\ell) - \varphi^{(L-\ell+1)}(\hat{\nu}^{\ell-1}) \right)^2 \mathbb{1}_{\mathcal{E}} \,\middle|\, \mathcal{F}^{\ell-1} \right] + \pi^2 \mathbb{E}\left[ \mathbb{1}_{\mathcal{E}^c} \,\middle|\, \mathcal{F}^{\ell-1} \right]$$

$$\leq \mathbb{E}\left[ \left( 2C\sqrt{\frac{d \log n}{n}}\left| \frac{\hat{\nu}^{\ell-1}}{1 + (c_0/8)(L-\ell)\hat{\nu}^{\ell-1}} \right| + 2C'n^{-c_1 d/2} \right)^2 \,\middle|\, \mathcal{F}^{\ell-1} \right] + C'''n^{-c_1 d}$$

$$\leq \left( 2C\sqrt{\frac{d \log n}{n}}\left| \frac{\hat{\nu}^{\ell-1}}{1 + (c_0/8)(L-\ell)\hat{\nu}^{\ell-1}} \right| + 2C'n^{-c_1 d/2} \right)^2 + C'''n^{-c_1 d}$$

where the first inequality uses the triangle inequality to obtain $\left(\varphi^{(L-\ell)}(\hat{\nu}^\ell) - \varphi^{(L-\ell+1)}(\hat{\nu}^{\ell-1})\right)^2 \leq \pi^2$; the second inequality applies the definition of $\mathcal{E}$, uses nonnegativity to drop the indicator in the first term, and applies the conditional measure bound on $\mathcal{E}^c$; and the final inequality integrates. Using the fact that our previous choices of large $n$ imply $2C\sqrt{d\log n/n} \leq 1/4$, and that $\left|\frac{\hat{\nu}^{\ell-1}}{1+(c_0/8)(L-\ell)\hat{\nu}^{\ell-1}}\right| \leq \pi$, we can distribute the square in this final bound and worst-case constants to obtain

$$\mathbb{E}\left[\left(\varphi^{(L-\ell)}(\hat{\nu}^\ell) - \varphi^{(L-\ell+1)}(\hat{\nu}^{\ell-1})\right)^2 \,\bigg|\, \mathcal{F}^{\ell-1}\right]$$

$$\leq 4C^2\frac{d\log n}{n}\left(\frac{\hat{\nu}^{\ell-1}}{1+(c_0/8)(L-\ell)\hat{\nu}^{\ell-1}}\right)^2 + C''''n^{-c_1d/2},$$

as claimed. $\qquad\square$

**Lemma D.26.** *Let $X_1,\ldots,X_L$ be independent chi-squared random variables, having respectively $d_1,\ldots,d_L$ degrees of freedom. Write $d_{\min} = \min_{i\in L} d_i$ and let $\xi_i = \frac{1}{d_i}X_i$. Then there are absolute constants $c,C > 0$ and an absolute constant $0 < K \leq \frac{1}{4}$ such that for any $0 < t \leq K$, one has*

$$\mathbb{P}\left[\left|-1+\prod_{i=1}^L\xi_i\right| > t\right] \leq CLe^{-cd_{\min}t^2/L}.$$

*In particular, there are absolute constants $C',C'' > 0$ and an absolute constant $K' > 0$ such that for any $d > 0$, if $d_{\min} \geq K'dL$ then one has*

$$\mathbb{P}\left[\left|-1+\prod_{i=1}^L\xi_i\right| > C'\sqrt{\frac{dL}{d_{\min}}}\right] \leq C''Le^{-d}.$$

*Proof.* For any $t \geq 0$, we have by the AM-GM inequality

$$\mathbb{P}\left[\prod_{i=1}^L\xi_i > 1+t\right] = \mathbb{P}\left[\left(\prod_{i=1}^L\xi_i\right)^{1/L} > (1+t)^{1/L}\right] \leq \mathbb{P}\left[\frac{1}{L}\sum_{i=1}^L\xi_i > (1+t)^{1/L}\right].$$

By convexity of the exponential, we have $(1+t)^{1/L} \geq 1 + \frac{1}{L}\log(1+t)$, and by concavity of the logarithm we have $\log(1+t) \geq t\log 2$ if $t \leq 1$. This implies

$$\mathbb{P}\left[\prod_{i=1}^L\xi_i > 1+t\right] \leq \mathbb{P}\left[-L+\sum_{i=1}^L\xi_i > Kt,\right],$$

where $K = \log(2)$. Decomposing each $X_i$ into a sum of $d_i$ i.i.d. squared gaussians and applying Lemma G.2, we obtain

$$\mathbb{P}\left[\left|-L+\sum_{i=1}^L\xi_i\right| > t\right] \leq 2\exp\left(-c\min\left\{\frac{t^2}{\sum_{i=1}^L\frac{C^2}{d_i}}, \frac{t}{C\max_i\frac{1}{d_i}}\right\}\right)$$

$$\leq 2\exp\left(-c'd_{\min}\min\{t^2/CL, t\}\right) \tag{D.142}$$

$$\leq 2\exp\left(-c''\frac{d_{\min}t^2}{L}\right),$$

where the last inequality holds provided $t \leq CL$, where $C > 0$ is an absolute constant. Thus, as long as $t \leq CL/K$, we have suitable control of the upper tail of the product $\prod_i\xi_i$. For the lower tail, writing $\log(0) = -\infty$, we have for any $0 \leq t < 1$

$$\mathbb{P}\left[\prod_{i=1}^L\xi_i < 1-t\right] = \mathbb{P}\left[\sum_{i=1}^L\log\xi_i < \log(1-t)\right] \leq \mathbb{P}\left[\sum_{i=1}^L\log\xi_i < -t\right],$$

where the inequality uses concavity of $t \mapsto \log(1-t)$. By Lemma G.2, we have for each $i \in [L]$ and every $0 \leq t \leq C$ (where $C > 0$ is an absolute constant)

$$\mathbb{P}[|\xi_i - 1| < t] \leq 2e^{-cd_it^2},$$

so that by a union bound and for $t \leq C\sqrt{L}$, we have with probability at least $1 - 2Le^{-cd_{\min}t^2/L}$ that $1 - t/\sqrt{L} \leq \xi_i \leq 1 + t/\sqrt{L}$ for every $i \in [L]$. Meanwhile, Taylor expansion of the smooth function $x \mapsto \log x$ in a neighborhood of 1 gives

$$\log x = (x - 1) - \frac{1}{2k^2}(x - 1)^2,$$

where $k$ is a number lying between 1 and $x$. In particular, if $x \geq \frac{1}{2}$ we have $\log x \geq (x - 1) - 2(x - 1)^2$, whence for $t \leq \min\{C\sqrt{L}, \frac{1}{2}\}$

$$\mathbb{P}\left[\prod_{i=1}^{L} \xi_i < 1 - t\right] \leq 2Le^{-cd_{\min}t^2/L} + \mathbb{P}\left[-L + \sum_{i=1}^{L} \xi_i < -t + 2t^2\right]$$

$$\leq 2Le^{-cd_{\min}t^2/L} + \mathbb{P}\left[-L + \sum_{i=1}^{L} \xi_i < -\frac{t}{2}\right],$$

where the final inequality requires in addition $t \leq \frac{1}{4}$. An application of (D.142) then yields the claimed lower tail provided $t \leq CL$, which establishes the first claim. For the second claim, we consider the choice $t = \sqrt{dL/cd_{\min}}$, for which we have $t \leq K$ whenever $d_{\min} \geq dL/cK^2$, and $cd_{\min}t^2/L = d$. $\qquad\square$

**Lemma D.27.** *Let $X_1, \ldots, X_L$ be independent* $\mathrm{Binom}(n, \frac{1}{2})$ *random variables, and write $\xi_i = \frac{2}{n}X_i$. Then for any $0 < t \leq \frac{1}{4}$, one has*

$$\mathbb{P}\left[\left|-1 + \prod_{i=1}^{L} \xi_i\right| > t\right] \leq 4Le^{-nt^2/8L}.$$

*In particular, for any $d > 0$, if $n \geq 128dL$ then one has*

$$\mathbb{P}\left[\left|-1 + \prod_{i=1}^{L} \xi_i\right| > 4\sqrt{\frac{dL}{n}}\right] \leq 4Le^{-d}.$$

*Proof.* The proof is very similar to that of Lemma D.26. For any $t \geq 0$, we have by the AM-GM inequality

$$\mathbb{P}\left[\prod_{i=1}^{L} \xi_i > 1 + t\right] = \mathbb{P}\left[\left(\prod_{i=1}^{L} \xi_i\right)^{1/L} > (1 + t)^{1/L}\right] \leq \mathbb{P}\left[\frac{1}{L}\sum_{i=1}^{L} \xi_i > (1 + t)^{1/L}\right].$$

By convexity of the exponential, we have $(1 + t)^{1/L} \geq 1 + \frac{1}{L}\log(1 + t)$, and by concavity of the logarithm we have $\log(1 + t) \geq t \log 2$ if $t \leq 1$. This implies

$$\mathbb{P}\left[\prod_{i=1}^{L} \xi_i > 1 + t\right] \leq \mathbb{P}\left[-L + \sum_{i=1}^{L} \xi_i > Kt,\right],$$

where $K = \log(2)$. Decomposing each $X_i$ into a sum of $n$ i.i.d. $\mathrm{Bern}(\frac{1}{2})$ random variables and applying Lemma G.1 twice, we obtain

$$\mathbb{P}\left[\left|-L + \sum_{i=1}^{L} \xi_i\right| > t\right] \leq 2e^{-nt^2/2L}. \tag{D.143}$$

This gives suitable control of the upper tail of the product $\prod_i \xi_i$. For the lower tail, writing $\log(0) = -\infty$, we have for any $0 \leq t < 1$

$$\mathbb{P}\left[\prod_{i=1}^{L} \xi_i < 1 - t\right] = \mathbb{P}\left[\sum_{i=1}^{L} \log \xi_i < \log(1 - t)\right] \leq \mathbb{P}\left[\sum_{i=1}^{L} \log \xi_i < -t\right],$$

where the inequality uses concavity of $t \mapsto \log(1 - t)$. By Lemma G.1, we have for each $i \in [L]$

$$\mathbb{P}[|\xi_i - 1| < t] \leq 2e^{-nt^2/2},$$

so that by a union bound, we have that $1 - t/\sqrt{L} \le \xi_i \le 1 + t/\sqrt{L}$ for every $i \in [L]$ with probability at least $1 - 2Le^{-nt^2/2L}$. Meanwhile, Taylor expansion of the smooth function $x \mapsto \log x$ in a neighborhood of 1 gives

$$\log x = (x - 1) - \frac{1}{2k^2}(x - 1)^2,$$

where $k$ is a number lying between 1 and $x$. In particular, if $x \ge \frac{1}{2}$ we have $\log x \ge (x-1) - 2(x-1)^2$, whence for $t \le 1/2$

$$\mathbb{P}\left[\prod_{i=1}^{L} \xi_i < 1 - t\right] \le 2Le^{-nt^2/2L} + \mathbb{P}\left[-L + \sum_{i=1}^{L} \xi_i < -t + 2t^2\right]$$

$$\le 2Le^{-nt^2/2L} + \mathbb{P}\left[-L + \sum_{i=1}^{L} \xi_i < -\frac{t}{2}\right],$$

where the final inequality requires in addition $t \le 1/4$. An application of (D.143) then yields the claimed lower tail, which establishes the first claim. For the second claim, we consider the choice $t = \sqrt{8dL/n}$, for which we have $t \le \frac{1}{4}$ whenever $n \ge 128dL$, and $nt^2/8L = d$. $\qquad\square$

**Lemma D.28.** *For $1 \le \ell' < \ell \le L - 1$ define events*

$$\tilde{\mathcal{E}}_B^{\ell:\ell'} = \left\{\left\|\boldsymbol{B}_{\boldsymbol{x}\boldsymbol{x}'}^{\ell-1:\ell'}\right\|_F^2 \le C^2 n(\ell - \ell')\right\} \quad\cap\quad \left\{\left\|\boldsymbol{B}_{\boldsymbol{x}\boldsymbol{x}'}^{\ell-1:\ell'}\right\| \le C(\ell - \ell')\right\}$$

$$\cap\quad \left\{\operatorname{tr}\left[\boldsymbol{B}_{\boldsymbol{x}\boldsymbol{x}'}^{\ell-1:\ell'}\right] \le Cn\right\}$$

*and*

$$\mathcal{E}_B^{\ell:\ell'} = \left\{\left\|\boldsymbol{\alpha}^{\ell-1}(\boldsymbol{x})\right\|_2 \left\|\boldsymbol{\alpha}^{\ell-1}(\boldsymbol{x}')\right\|_2 > 0\right\} \quad\cap\quad \left\{\left|\varphi^{(\ell-1)}(\nu) - \nu^{\ell-1}\right| \le C\sqrt{\frac{d^3 \log^3 n}{n\ell}}\right\}$$

$$\cap\quad \tilde{\mathcal{E}}_B^{\ell:\ell'}.$$

*If $n, L$ satisfy the assumptions of corollary D.17 then*

$$\mathbb{P}\left[\tilde{\mathcal{E}}_B^{\ell:\ell'}\right] \ge 1 - C'n(\ell - \ell')^2 e^{-c\frac{n}{\ell-\ell'}}.$$

*If $n, L$ additionally satisfy the conditions of lemmas D.3, E.16 and $n \ge C''L\log(n)(\log(L) + d)$, then*

$$\mathbb{P}\left[\mathcal{E}_B^{\ell:\ell'}\right] \ge 1 - C'n^{-cd}.$$

*where $c, C, C', C''$ are absolute constants.*

*Proof.* Since

$$\operatorname{tr}\left[\boldsymbol{B}_{\boldsymbol{x}\boldsymbol{x}'}^{\ell-1:\ell'}\right] = \sum_{i=1}^{n} \boldsymbol{e}_i^* \boldsymbol{\Gamma}^{\ell-1:\ell'+2}(\boldsymbol{x}) \boldsymbol{P}_{I_{\ell'+1}(\boldsymbol{x})} \boldsymbol{P}_{I_{\ell'+1}(\boldsymbol{x}')} \boldsymbol{\Gamma}^{\ell-1:\ell'+2*}(\boldsymbol{x}') \boldsymbol{e}_i,$$

applying corollary D.17 $2n$ times gives

$$\mathbb{P}\left[\bigcap_{\boldsymbol{z} \in \{\boldsymbol{x}, \boldsymbol{x}'\}, i \in [n]} \left\{\left\|\boldsymbol{\Gamma}^{\ell-1:\ell'+2}(\boldsymbol{z}) \boldsymbol{e}_i\right\|_2 \le \sqrt{C}\right\}\right] \ge 1 - C'n(\ell - \ell')^2 e^{-c\frac{n}{\ell-\ell'}}$$

$$\Rightarrow \mathbb{P}\left[\operatorname{tr}\left[\boldsymbol{B}_{\boldsymbol{x}\boldsymbol{x}'}^{\ell-1:\ell'}\right] \le Cn\right] \ge 1 - C'n(\ell - \ell')^2 e^{-c\frac{n}{\ell-\ell'}}.$$

With the same probability we also have

$$\max_{\boldsymbol{z} \in \{\boldsymbol{x}, \boldsymbol{x}'\}} \left\|\boldsymbol{\Gamma}^{\ell-1:\ell'+2}(\boldsymbol{z})\right\|_F^2 = \max_{\boldsymbol{z} \in \{\boldsymbol{x}, \boldsymbol{x}'\}} \operatorname{tr}\left[\boldsymbol{\Gamma}^{\ell-1:\ell'+2*}(\boldsymbol{z}) \boldsymbol{\Gamma}^{\ell-1:\ell'+2}(\boldsymbol{z})\right] \le Cn$$

and

$$\mathbb{P}\left[\max_{\boldsymbol{z}\in\{\boldsymbol{x},\boldsymbol{x}'\}}\left\|\boldsymbol{\Gamma}^{\ell-1:\ell'+2}(\boldsymbol{z})\right\| \le \sqrt{C(\ell-\ell')}\right] \ge 1 - C''(\ell-\ell')^3 e^{-c\frac{n}{\ell-\ell'}}$$

from which it follows that

$$\mathbb{P}\left[\left\|\boldsymbol{B}_{\boldsymbol{x}\boldsymbol{x}'}^{\ell-1:\ell'}\right\| \le C(\ell-\ell')\right] \ge 1 - C''(\ell-\ell')^3 e^{-c\frac{n}{\ell-\ell'}}$$

and

$$\mathbb{P}\left[\left\|\boldsymbol{B}_{\boldsymbol{x}\boldsymbol{x}'}^{\ell-1:\ell'}\right\|_F^2 \le C^2 n(\ell-\ell')\right]$$

$$\ge \mathbb{P}\left[\max_{\boldsymbol{z}\in\{\boldsymbol{x},\boldsymbol{x}'\}}\left\|\boldsymbol{\Gamma}^{\ell-1:\ell'+2}(\boldsymbol{z})\right\|^2 \max_{\boldsymbol{z}\in\{\boldsymbol{x},\boldsymbol{x}'\}}\left\|\boldsymbol{\Gamma}^{\ell-1:\ell'+2}(\boldsymbol{x})\right\|_F^2 \le C^2 n(\ell-\ell')\right]$$

$$\ge 1 - C''(\ell-\ell')^3 e^{-c\frac{n}{\ell-\ell'}} - C'n(\ell-\ell')^2 e^{-c\frac{n}{\ell-\ell'}}$$

$$\ge 1 - C'''n(\ell-\ell')^2 e^{-c\frac{n}{\ell-\ell'}}.$$

It follows that $\tilde{\mathcal{E}}_B^{\ell:\ell'}$ holds with the same probability.

From lemma E.16,

$$\mathbb{P}\left[\left\|\boldsymbol{\alpha}^{\ell-1}(\boldsymbol{x})\right\|_2 \left\|\boldsymbol{\alpha}^{\ell-1}(\boldsymbol{x}')\right\|_2 > 0 \cap \nu^{\ell-1} = \widehat{\nu}^{\ell-1}\right] \ge 1 - C'\ell e^{-cn}$$

for some constants $c, C'$. Here $\widehat{\nu}^{\ell-1}$ is the auxiliary angle process defined in (D.2). Using D.3, we obtain

$$\mathbb{P}\left[\left|\varphi^{(\ell-1)}(\nu) - \nu^{\ell-1}\right| \le C\sqrt{\frac{d^3 \log^3 n}{n\ell}}\right] \ge \mathbb{P}\left[\left|\varphi^{(\ell-1)}(\nu) - \nu^{\ell-1}\right| \le C\sqrt{\frac{d^3 \log^3 n}{n\ell}}\bigg|\mathcal{E}\right] + \mathbb{P}\left[\mathcal{E}^c\right]$$

$$\ge 1 - C'''\ell e^{-cn} - C''n^{-cd} \ge 1 - C'n^{-cd}$$

for an appropriate choice of $c, C'$.

We conclude that

$$\mathbb{P}\left[\mathcal{E}_B^{\ell:\ell'}\right] \ge 1 - C'e^{-c'n} - C''n\ell^2 e^{-c''\frac{n}{\ell-\ell'}} - C'''n^{-c'''d}$$

$$\ge 1 - \tilde{C}n^{-\tilde{c}d}$$

for appropriately chosen constants, where we used $n \ge C''''\ell \log(n)(\log(\ell) + d)$. $\qquad\square$

**Lemma D.29.** *For $\overline{\Delta}_\ell$ defined in* (D.34) *and $\mathcal{E}_B^{\ell:\ell'}$ defined in lemma D.28 we have*

$$\mathbb{P}\left[\left|\mathbb{1}_{\mathcal{E}_B^{\ell:\ell'}}\overline{\Delta}_\ell\right| > C\sqrt{d\ell}\bigg|\mathcal{F}^{\ell-1}\right] \underset{a.s.}{\le} C'e^{-cd}.$$

*for some constants $c, C, C'$.*

*Proof.*

$$\mathbb{1}_{\mathcal{E}_B^{\ell:\ell'}}\overline{\Delta}_\ell = \mathbb{1}_{\mathcal{E}_B^{\ell:\ell'}}(\Delta_\ell - \mathbb{E}\Delta_\ell|\mathcal{F}^{\ell-1})$$

$$= \mathbb{1}_{\mathcal{E}_B^{\ell:\ell'}}\prod_{i=\ell}^{L-1}\left(1 - \frac{\varphi^{(i)}(\nu)}{\pi}\right)\left(\text{tr}\left[\boldsymbol{B}_{\boldsymbol{x}\boldsymbol{x}'}^{\ell:\ell'}\right] - \mathbb{E}_{\boldsymbol{W}^\ell}\text{tr}\left[\boldsymbol{B}_{\boldsymbol{x}\boldsymbol{x}'}^{\ell:\ell'}\right]\right),$$

and denoting $\boldsymbol{P}_{\boldsymbol{x}\boldsymbol{x}'}^\ell = \boldsymbol{P}_{I_\ell(\boldsymbol{x})}\boldsymbol{P}_{I_\ell(\boldsymbol{x}')}$ we have

$$\text{tr}\left[\boldsymbol{B}_{\boldsymbol{x}\boldsymbol{x}'}^{\ell:\ell'}\right] - \mathbb{E}_{\boldsymbol{W}^\ell}\text{tr}\left[\boldsymbol{B}_{\boldsymbol{x}\boldsymbol{x}'}^{\ell:\ell'}\right] = \text{tr}\left[\boldsymbol{B}_{\boldsymbol{x}\boldsymbol{x}'}^{\ell-1:\ell'}\left(\boldsymbol{W}^{\ell*}\boldsymbol{P}_{\boldsymbol{x}\boldsymbol{x}'}^\ell\boldsymbol{W}^\ell - \mathbb{E}_{\boldsymbol{W}^\ell}\left[\boldsymbol{W}^{\ell*}\boldsymbol{P}_{\boldsymbol{x}\boldsymbol{x}'}^\ell\boldsymbol{W}^\ell\right]\right)\right]$$

Defining $S^\ell = \text{span}\{\boldsymbol{\alpha}^\ell(\boldsymbol{x}), \boldsymbol{\alpha}^\ell(\boldsymbol{x}')\}$ we decompose $\boldsymbol{W}^\ell$ into a sum of two independent terms as

$$\boldsymbol{W}^\ell = \boldsymbol{W}^\ell\boldsymbol{P}_{S^{\ell-1}} + \boldsymbol{W}^\ell\boldsymbol{P}_{S^{\ell-1\perp}} \equiv \boldsymbol{G}^\ell + \boldsymbol{H}^\ell.$$

Note that each $\boldsymbol{H}^\ell$ is independent of every other random variable in the problem conditioned on the features. $\mathbb{1}_{\mathcal{E}_B^{\ell:\ell'}} \operatorname{tr}\left[\boldsymbol{B}_{\boldsymbol{x}\boldsymbol{x}'}^{\ell:\ell'}\right]$ thus decomposes into a sum of four terms, which we proceed to consider individually and show that they concentrate.

The all $\boldsymbol{G}^\ell$ term is

$$\mathbb{1}_{\mathcal{E}_B^{\ell:\ell'}} \operatorname{tr}\left[\boldsymbol{B}_{\boldsymbol{x}\boldsymbol{x}'}^{\ell-1:\ell'} \boldsymbol{G}^{\ell*} \boldsymbol{P}_{\boldsymbol{x}\boldsymbol{x}'}^\ell \boldsymbol{G}^\ell\right] = \mathbb{1}_{\mathcal{E}_B^{\ell:\ell'}} \sum_{i,j=1}^{\dim S^{\ell-1}} \boldsymbol{u}_j^{\ell-1*} \boldsymbol{B}_{\boldsymbol{x}\boldsymbol{x}'}^{\ell-1:\ell'} \boldsymbol{u}_i^{\ell-1} \boldsymbol{u}_i^{\ell-1*} \boldsymbol{W}^{\ell*} \boldsymbol{P}_{\boldsymbol{x}\boldsymbol{x}'}^\ell \boldsymbol{W}^\ell \boldsymbol{u}_j^{\ell-1}$$

where $\{\boldsymbol{u}_i^{\ell-1}\}$ is an orthonormal basis of $S^{\ell-1}$. If $\boldsymbol{\alpha}^{\ell-1}(\boldsymbol{x}) \neq \boldsymbol{\alpha}^{\ell-1}(\boldsymbol{x}')$ we choose

$$(\boldsymbol{u}_1^{\ell-1}, \boldsymbol{u}_2^{\ell-1}) = \left(\frac{\boldsymbol{\alpha}^{\ell-1}(\boldsymbol{x})}{\|\boldsymbol{\alpha}^{\ell-1}(\boldsymbol{x})\|_2}, \frac{\boldsymbol{P}_{\boldsymbol{\alpha}^{\ell-1}(\boldsymbol{x})\perp} \boldsymbol{\alpha}^{\ell-1}(\boldsymbol{x}')}{\|\boldsymbol{P}_{\boldsymbol{\alpha}^{\ell-1}(\boldsymbol{x})\perp} \boldsymbol{\alpha}^{\ell-1}(\boldsymbol{x}')\|_2}\right),$$

(which are well-defined on $\mathcal{E}_B^{\ell:\ell'}$).Using rotational invariance of the Gaussian distribution, we have

$$\boldsymbol{u}_i^* \boldsymbol{W}^{\ell-1*} \boldsymbol{P}_{\boldsymbol{x}\boldsymbol{x}'}^{\ell-1+1} \boldsymbol{W}^{\ell-1} \boldsymbol{u}_j$$
$$\stackrel{d}{=} \boldsymbol{u}_i^{\ell-1*} \boldsymbol{R}^* \boldsymbol{W}^{\ell-1*} \boldsymbol{P}_{\boldsymbol{W}^{\ell-1} \boldsymbol{R}\boldsymbol{\alpha}^{\ell-1}(\boldsymbol{x}')>\boldsymbol{0}} \boldsymbol{P}_{\boldsymbol{W}^{\ell-1} \boldsymbol{R}\boldsymbol{\alpha}^{\ell-1}(\boldsymbol{x})>\boldsymbol{0}} \boldsymbol{W}^{\ell-1} \boldsymbol{R} \boldsymbol{u}_j^{\ell-1}$$
$$\stackrel{d}{=} \left\langle \boldsymbol{P}_{\boldsymbol{g}_1 \cos \nu^{\ell-1} + \boldsymbol{g}_2 \sin \nu^{\ell-1} > \boldsymbol{0}} \boldsymbol{g}_i, \boldsymbol{P}_{\boldsymbol{g}_1 > \boldsymbol{0}} \boldsymbol{g}_j \right\rangle$$

where $\boldsymbol{g}_i \sim_{\text{iid}} \mathcal{N}(0, \frac{2}{n}\boldsymbol{I})$. If $\boldsymbol{\alpha}^{\ell-1}(\boldsymbol{x}) = \boldsymbol{\alpha}^{\ell-1}(\boldsymbol{x}')$ then $\dim S^{\ell-1} = 1$ and we simply choose $\boldsymbol{u}_1^{\ell-1} = \frac{\boldsymbol{\alpha}^{\ell-1}(\boldsymbol{x})}{\|\boldsymbol{\alpha}^{\ell-1}(\boldsymbol{x})\|_2}$ and end up with an identical expression, with $\nu^{\ell-1} = 0$. Since $\boldsymbol{P}_{\boldsymbol{g}_1 > \boldsymbol{0}} \boldsymbol{g}_j$ and $\boldsymbol{P}_{\boldsymbol{g}_1 \cos \nu^{\ell-1} + \boldsymbol{g}_2 \sin \nu^{\ell-1} > \boldsymbol{0}} \boldsymbol{g}_i$ are vectors of independent sub-Gaussian random variables with sub-Gaussian norm bounded by $\sqrt{\frac{C}{n}}$, their inner product is a sum of independent sub-exponential variables with sub-exponential norm bounded by $\frac{C}{n}$ for some constant $C$. Momentarily abbreviating $\tilde{\boldsymbol{v}} = \boldsymbol{g}_1 \cos \nu^{\ell-1} + \boldsymbol{g}_2 \sin \nu^{\ell-1}$, Bernstein's inequality then gives

$$\mathbb{P}\left[\left|\langle \boldsymbol{P}_{\tilde{\boldsymbol{v}}>\boldsymbol{0}} \boldsymbol{g}_i, \boldsymbol{P}_{\boldsymbol{g}_1>\boldsymbol{0}} \boldsymbol{g}_j\rangle - \mathop{\mathbb{E}}_{\boldsymbol{g}_1,\boldsymbol{g}_2} \langle \boldsymbol{P}_{\tilde{\boldsymbol{v}}>\boldsymbol{0}} \boldsymbol{g}_i, \boldsymbol{P}_{\boldsymbol{g}_1>\boldsymbol{0}} \boldsymbol{g}_j\rangle\right| > \sqrt{\frac{d}{n}} \Bigg| \mathcal{F}^{\ell-1}\right] \underset{a.s.}{\leq} 2e^{-cd} \qquad \text{(D.144)}$$

for some constant $c$. Since on $\mathcal{E}_B^{\ell:\ell'}$, $\left\|\boldsymbol{B}_{\boldsymbol{x}\boldsymbol{x}'}^{\ell-1:\ell'}\right\| \leq C\ell$, we obtain

$$\left|\mathbb{1}_{\mathcal{E}_B^{\ell:\ell'}} \operatorname{tr}\left[\boldsymbol{B}_{\boldsymbol{x}\boldsymbol{x}'}^{\ell-1:\ell'} \boldsymbol{G}^{\ell*} \boldsymbol{P}_{\boldsymbol{x}\boldsymbol{x}'}^\ell \boldsymbol{G}^\ell\right]\right| \leq C\ell \sum_{i,j=1}^{\dim S^{\ell-1}} \boldsymbol{u}_i^{\ell-1*} \boldsymbol{W}^{\ell*} \boldsymbol{P}_{\boldsymbol{x}\boldsymbol{x}'}^\ell \boldsymbol{W}^\ell \boldsymbol{u}_j^{\ell-1}$$

almost surely and thus

$$\mathbb{P}\left[\left|\mathbb{1}_{\mathcal{E}_B^{\ell:\ell'}} \operatorname{tr}\left[\boldsymbol{B}_{\boldsymbol{x}\boldsymbol{x}'}^{\ell-1:\ell'} \boldsymbol{G}^{\ell*} \boldsymbol{P}_{\boldsymbol{x}\boldsymbol{x}'}^\ell \boldsymbol{G}^\ell\right]\right| > C'\ell \sqrt{\frac{d}{n}} \Bigg| \mathcal{F}^{\ell-1}\right] \underset{a.s.}{\leq} 2e^{-cd}$$

for some $C'$, and hence

$$\mathbb{P}\left[\left|\begin{array}{l} \mathbb{1}_{\mathcal{E}_B^{\ell:\ell'}} \operatorname{tr}\left[\boldsymbol{B}_{\boldsymbol{x}\boldsymbol{x}'}^{\ell-1:\ell'} \boldsymbol{G}^{\ell*} \boldsymbol{P}_{\boldsymbol{x}\boldsymbol{x}'}^\ell \boldsymbol{G}^\ell\right] \\ - \mathop{\mathbb{E}}_{\boldsymbol{W}^\ell} \mathbb{1}_{\mathcal{E}_B^{\ell:\ell'}} \operatorname{tr}\left[\boldsymbol{B}_{\boldsymbol{x}\boldsymbol{x}'}^{\ell-1:\ell'} \boldsymbol{G}^{\ell*} \boldsymbol{P}_{\boldsymbol{x}\boldsymbol{x}'}^\ell \boldsymbol{G}^\ell\right] \end{array}\right| \leq 2C'\sqrt{\ell d} \Bigg| \mathcal{F}^{\ell-1}\right] \underset{a.s.}{\leq} 2e^{-c\frac{n}{\ell}}. \qquad \text{(D.145)}$$

$\operatorname{tr}\left[\boldsymbol{B}_{\boldsymbol{x}\boldsymbol{x}'}^{\ell:\ell'}\right]$ also contains the terms

$$\operatorname{tr}\left[\boldsymbol{B}_{\boldsymbol{x}\boldsymbol{x}'}^{\ell-1:\ell'} \boldsymbol{G}^{\ell*} \boldsymbol{P}_{\boldsymbol{x}\boldsymbol{x}'}^\ell \boldsymbol{H}^\ell\right] + \operatorname{tr}\left[\boldsymbol{B}_{\boldsymbol{x}\boldsymbol{x}'}^{\ell-1:\ell'} \boldsymbol{H}^{\ell*} \boldsymbol{P}_{\boldsymbol{x}\boldsymbol{x}'}^\ell \boldsymbol{G}^\ell\right].$$

Considering the first of these (since the second can be treated in an identical fashion), we recall that $\boldsymbol{H}^\ell$ is independent of all the other random variables in the problem conditioned on the features, and we thus have

$$\mathbb{1}_{\mathcal{E}_B^{\ell:\ell'}} \operatorname{tr}\left[\boldsymbol{B}_{\boldsymbol{x}\boldsymbol{x}'}^{\ell-1:\ell'} \boldsymbol{G}^{\ell*} \boldsymbol{P}_{\boldsymbol{x}\boldsymbol{x}'}^\ell \boldsymbol{H}^\ell\right] \stackrel{d}{=} \mathbb{1}_{\mathcal{E}_B^{\ell:\ell'}} \operatorname{tr}\left[\boldsymbol{B}_{\boldsymbol{x}\boldsymbol{x}'}^{\ell-1:\ell'} \boldsymbol{G}^{\ell*} \boldsymbol{P}_{\boldsymbol{x}\boldsymbol{x}'}^\ell \tilde{\boldsymbol{W}}^\ell \boldsymbol{P}_{S^{\ell-1}\perp}\right]$$
$$= \sum_{i=1}^{\dim S^{\ell-1}} \mathbb{1}_{\mathcal{E}_B^{\ell:\ell'}} \boldsymbol{v}_i^{\ell*} \tilde{\boldsymbol{W}}^\ell \boldsymbol{w}_i^\ell$$

where $\tilde{\boldsymbol{W}}^\ell$ is an independent copy of $\boldsymbol{W}^\ell$, $\boldsymbol{v}_i^\ell = \boldsymbol{P}_{\boldsymbol{xx}'}^\ell \boldsymbol{u}_i^\ell$, $\boldsymbol{w}_i^\ell = \boldsymbol{P}_{S^{\ell-1}\perp} \boldsymbol{B}_{\boldsymbol{xx}'}^{\ell-1:\ell'} \boldsymbol{u}_i^\ell$. Hence conditioned on all the other variables, $\boldsymbol{v}_i^{\ell*} \tilde{\boldsymbol{W}}^\ell \boldsymbol{w}_i^\ell$ is a zero-mean Gaussian with variance $\frac{2\|\boldsymbol{v}_i^\ell\|_2^2 \|\boldsymbol{w}_i^\ell\|_2^2}{n} \mathbb{1}_{\mathcal{E}_B^{\ell:\ell'}}$. Again from the bound on $\left\| \boldsymbol{B}_{\boldsymbol{xx}'}^{\ell-1:\ell'} \right\|$ implied by $\mathcal{E}_B^{\ell:\ell'}$, we have

$$\frac{2 \left\| \boldsymbol{v}_i^\ell \right\|_2^2 \left\| \boldsymbol{w}_i^\ell \right\|_2^2}{n} \mathbb{1}_{\mathcal{E}_B^{\ell:\ell'}} \leq \frac{C\ell^2}{n}$$

almost surely. Noting that $\underset{\boldsymbol{W}^\ell}{\mathbb{E}} \operatorname{tr} \left[ \boldsymbol{B}_{\boldsymbol{xx}'}^{\ell-1:\ell'} \boldsymbol{G}^{\ell*} \boldsymbol{P}_{\boldsymbol{xx}'}^\ell \boldsymbol{H}^\ell \right] = \underset{\boldsymbol{G}^\ell}{\mathbb{E}} \underset{\tilde{\boldsymbol{W}}^\ell}{\mathbb{E}} \operatorname{tr} \left[ \sum_{i=1}^{\dim S^{\ell-1}} \boldsymbol{v}_i^{\ell*} \tilde{\boldsymbol{W}}^\ell \boldsymbol{w}_i^\ell \right] = 0$, a Gaussian tail bound gives

$$\mathbb{P} \left[ \left| \mathbb{1}_{\mathcal{E}_B^{\ell:\ell'}} \operatorname{tr} \left[ \boldsymbol{B}_{\boldsymbol{xx}'}^{\ell-1:\ell'} \boldsymbol{G}^{\ell*} \boldsymbol{P}_{\boldsymbol{xx}'}^\ell \boldsymbol{H}^\ell \right] - \underset{\boldsymbol{W}^\ell}{\mathbb{E}} \mathbb{1}_{\mathcal{E}_B^{\ell:\ell'}} \operatorname{tr} \left[ \boldsymbol{B}_{\boldsymbol{xx}'}^{\ell-1:\ell'} \boldsymbol{G}^{\ell*} \boldsymbol{P}_{\boldsymbol{xx}'}^\ell \boldsymbol{H}^\ell \right] \right| > \sqrt{\ell} \, \middle| \, \mathcal{F}^{\ell-1} \right] \tag{D.146}$$
$$\underset{a.s.}{\leq} 2 e^{-c\frac{n}{\ell}}.$$

The final term in $\operatorname{tr} \left[ \boldsymbol{B}_{\boldsymbol{xx}'}^{\ell:\ell'} \right]$ is

$$\operatorname{tr} \left[ \boldsymbol{B}_{\boldsymbol{xx}'}^{\ell-1:\ell'} \boldsymbol{H}^{\ell*} \boldsymbol{P}_{\boldsymbol{xx}'}^\ell \boldsymbol{H}^\ell \right] \stackrel{d}{=} \operatorname{tr} \left[ \boldsymbol{B}_{\boldsymbol{xx}'}^{\ell-1:\ell'} \boldsymbol{P}_{S^{\ell-1}\perp} \tilde{\boldsymbol{W}}^{\ell*} \boldsymbol{P}_{\boldsymbol{xx}'}^\ell \tilde{\boldsymbol{W}}^\ell \boldsymbol{P}_{S^{\ell-1}\perp} \right].$$

Due to the independence of $\tilde{\boldsymbol{W}}^\ell$ from the remaining random variables, this is simply a Gaussian chaos in $n^2$ variables. The Hanson-Wright inequality (lemma G.4) gives

$$\mathbb{P} \left[ \left| \mathbb{1}_{\mathcal{E}_B^{\ell:\ell'}} \operatorname{tr} \left[ \boldsymbol{B}_{\boldsymbol{xx}'}^{\ell-1:\ell'} \boldsymbol{H}^{\ell*} \boldsymbol{P}_{\boldsymbol{xx}'}^\ell \boldsymbol{H}^\ell \right] - \underset{\boldsymbol{W}^\ell}{\mathbb{E}} \mathbb{1}_{\mathcal{E}_B^{\ell:\ell'}} \operatorname{tr} \left[ \boldsymbol{B}_{\boldsymbol{xx}'}^{\ell-1:\ell'} \boldsymbol{H}^{\ell*} \boldsymbol{P}_{\boldsymbol{xx}'}^\ell \boldsymbol{H}^\ell \right] \right| \geq t \, \middle| \, \mathcal{F}^{\ell-1} \right]$$

$$\underset{a.s.}{\leq} 2 \exp \left( -cnt \min \left\{ \frac{t}{\left\| \mathbb{1}_{\mathcal{E}_B^{\ell:\ell'}} \boldsymbol{P}_{S^{\ell-1}\perp} \boldsymbol{B}_{\boldsymbol{xx}'}^{\ell-1:\ell'} \boldsymbol{P}_{S^{\ell-1}\perp} \right\|_F^2}, \frac{1}{\left\| \mathbb{1}_{\mathcal{E}_B^{\ell:\ell'}} \boldsymbol{P}_{S^{\ell-1}\perp} \boldsymbol{B}_{\boldsymbol{xx}'}^{\ell-1:\ell'} \boldsymbol{P}_{S^{\ell-1}\perp} \right\|} \right\} \right)$$

$$\underset{a.s.}{\leq} 2 \exp \left( -c\frac{n}{\ell} t \min \left\{ \frac{t}{n}, 1 \right\} \right)$$

where in the last inequality we used the definition of $\mathcal{E}_B^{\ell:\ell'}$. It follows that

$$\mathbb{P} \left[ \left| \mathbb{1}_{\mathcal{E}_B^{\ell:\ell'}} \operatorname{tr} \left[ \boldsymbol{B}_{\boldsymbol{xx}'}^{\ell-1:\ell'} \boldsymbol{H}^{\ell*} \boldsymbol{P}_{\boldsymbol{xx}'}^\ell \boldsymbol{H}^\ell \right] - \underset{\boldsymbol{W}^\ell}{\mathbb{E}} \mathbb{1}_{\mathcal{E}_B^{\ell:\ell'}} \operatorname{tr} \left[ \boldsymbol{B}_{\boldsymbol{xx}'}^{\ell-1:\ell'} \boldsymbol{H}^{\ell*} \boldsymbol{P}_{\boldsymbol{xx}'}^\ell \boldsymbol{H}^\ell \right] \right| > 2\sqrt{d\ell} \, \middle| \, \mathcal{F}^{\ell-1} \right]$$

$$\underset{a.s.}{\leq} \begin{aligned} &\mathbb{P} \left[ \left| \mathbb{1}_{\mathcal{E}_B^{\ell:\ell'}} \operatorname{tr} \left[ \boldsymbol{B}_{\boldsymbol{xx}'}^{\ell-1:\ell'} \boldsymbol{H}^{\ell*} \boldsymbol{P}_{\boldsymbol{xx}'}^\ell \boldsymbol{H}^\ell \right] - \underset{\boldsymbol{H}^\ell}{\mathbb{E}} \mathbb{1}_{\mathcal{E}_B^{\ell:\ell'}} \operatorname{tr} \left[ \boldsymbol{B}_{\boldsymbol{xx}'}^{\ell-1:\ell'} \boldsymbol{H}^{\ell*} \boldsymbol{P}_{\boldsymbol{xx}'}^\ell \boldsymbol{H}^\ell \right] \right| > \sqrt{d\ell} \, \middle| \, \mathcal{F}^{\ell-1} \right] \\ &+ \mathbb{P} \left[ \left| \underset{\boldsymbol{H}^\ell}{\mathbb{E}} \mathbb{1}_{\mathcal{E}_B^{\ell:\ell'}} \operatorname{tr} \left[ \boldsymbol{B}_{\boldsymbol{xx}'}^{\ell-1:\ell'} \boldsymbol{H}^{\ell*} \boldsymbol{P}_{\boldsymbol{xx}'}^\ell \boldsymbol{H}^\ell \right] - \underset{\boldsymbol{W}^\ell}{\mathbb{E}} \mathbb{1}_{\mathcal{E}_B^{\ell:\ell'}} \operatorname{tr} \left[ \boldsymbol{B}_{\boldsymbol{xx}'}^{\ell-1:\ell'} \boldsymbol{H}^{\ell*} \boldsymbol{P}_{\boldsymbol{xx}'}^\ell \boldsymbol{H}^\ell \right] \right| > \sqrt{d\ell} \, \middle| \, \mathcal{F}^{\ell-1} \right] \end{aligned}$$

$$\underset{a.s.}{\leq} \begin{aligned} &\mathbb{P} \left[ \left| \mathbb{1}_{\mathcal{E}_B^{\ell:\ell'}} \operatorname{tr} \left[ \boldsymbol{B}_{\boldsymbol{xx}'}^{\ell-1:\ell'} \boldsymbol{H}^{\ell*} \boldsymbol{P}_{\boldsymbol{xx}'}^\ell \boldsymbol{H}^\ell \right] - \underset{\boldsymbol{H}^\ell}{\mathbb{E}} \mathbb{1}_{\mathcal{E}_B^{\ell:\ell'}} \operatorname{tr} \left[ \boldsymbol{B}_{\boldsymbol{xx}'}^{\ell-1:\ell'} \boldsymbol{H}^{\ell*} \boldsymbol{P}_{\boldsymbol{xx}'}^\ell \boldsymbol{H}^\ell \right] \right| > \sqrt{d\ell} \, \middle| \, \mathcal{F}^{\ell-1} \right] \\ &+ \mathbb{P} \left[ \mathbb{1}_{\mathcal{E}_B^{\ell:\ell'}} \left| \frac{2\operatorname{tr} \left[ \boldsymbol{P}_{S^{\ell-1}\perp} \boldsymbol{B}_{\boldsymbol{xx}'}^{\ell-1:\ell'} \boldsymbol{P}_{S^{\ell-1}\perp} \right]}{n} \right| \left| \operatorname{tr} \left[ \boldsymbol{P}_{\boldsymbol{xx}'}^\ell \right] - \underset{\boldsymbol{H}^\ell}{\mathbb{E}} \operatorname{tr} \left[ \boldsymbol{P}_{\boldsymbol{xx}'}^\ell \right] \right| > \sqrt{d\ell} \, \middle| \, \mathcal{F}^{\ell-1} \right] \end{aligned}$$

$$\underset{a.s.}{\leq} C e^{-cd} \tag{D.147}$$

where in the last inequality we used (D.144) and the properties of $\mathcal{E}_B^{\ell:\ell'}$. Collecting terms and using (D.145), (D.146), (D.147) we obtain

$$\mathbb{P} \left[ \mathbb{1}_{\mathcal{E}_B^{\ell:\ell'}} \left| \operatorname{tr} \left[ \boldsymbol{B}_{\boldsymbol{xx}'}^{\ell:\ell'} \right] - \underset{\boldsymbol{W}^\ell}{\mathbb{E}} \operatorname{tr} \left[ \boldsymbol{B}_{\boldsymbol{xx}'}^{\ell:\ell'} \right] \right| > C\sqrt{d\ell} \, \middle| \, \mathcal{F}^{\ell-1} \right] \underset{a.s.}{\leq} C' e^{-cd} \tag{D.148}$$

and hence

$$\mathbb{P}\left[\left|\mathbb{1}_{\mathcal{E}_B^{\ell:\ell'}}\overline{\Delta}_\ell\right| > C\sqrt{d\ell}\,\middle|\, \mathcal{F}^{\ell-1}\right]$$

$$= \mathbb{P}\left[\mathbb{1}_{\mathcal{E}_B^{\ell:\ell'}}\prod_{i=\ell}^{L-1}\left(1 - \frac{\varphi^{(i)}(\nu)}{\pi}\right)\left|\mathrm{tr}\left[\boldsymbol{B}_{\boldsymbol{x}\boldsymbol{x}'}^{\ell:\ell'}\right] - \underset{\boldsymbol{W}^\ell}{\mathbb{E}}\,\mathrm{tr}\left[\boldsymbol{B}_{\boldsymbol{x}\boldsymbol{x}'}^{\ell:\ell'}\right]\right| > C\sqrt{d\ell}\,\middle|\, \mathcal{F}^{\ell-1}\right]$$

$$\underset{a.s.}{\leq} C'e^{-cd}.$$

(D.149)

$\square$

**Lemma D.30.** *For $\boldsymbol{x} \in \mathbb{S}^{n_0-1}$ and $\ell \in [L]$, denote $I_\ell(\boldsymbol{x}) = \mathrm{supp}(\boldsymbol{\alpha}^\ell(\boldsymbol{x}) > \boldsymbol{0})$. If $n \geq K$ then*

$$\mathbb{P}\left[\min_\ell |I_\ell(\boldsymbol{x})| \geq \frac{n}{4}\right] \geq 1 - 2LCe^{-cn}$$

*and for any $0 \leq t \leq 1$*

$$\mathbb{P}\left[\prod_{\ell=1}^L \frac{2\,|I_\ell(\boldsymbol{x})|}{n} - 1 \geq t\right] \leq 2\exp\left(-c\frac{n}{L}t^2\right)$$

*where $c, c', C, K$ are absolute constants.*

*Proof.* Consider the activations at layer $\ell$. From lemma E.16, if $n \geq K$ we have

$$\mathbb{P}\left[\left\|\boldsymbol{\alpha}^{\ell-1}(\boldsymbol{x})\right\|_2 > 0\right] \geq 1 - Ce^{-cn}.$$

Rotational invariance of the Gaussian distribution gives $\boldsymbol{\alpha}^\ell(\boldsymbol{x}) = \left[\boldsymbol{W}^\ell\boldsymbol{\alpha}^{\ell-1}(\boldsymbol{x})\right]_+ \overset{d}{=} \left\|\boldsymbol{\alpha}^{\ell-1}(\boldsymbol{x})\right\|_2\left[\boldsymbol{W}_{(:,1)}^\ell\right]_+$. It follows that

$$\mathbb{E}\left[|I_\ell(\boldsymbol{x})|\,\middle|\,\left\|\boldsymbol{\alpha}^{\ell-1}(\boldsymbol{x})\right\|_2 > 0\right] = \mathbb{E}\left[\sum_{i=1}^n \mathbb{1}_{\alpha_i^\ell(\boldsymbol{x})>0}\,\middle|\,\left\|\boldsymbol{\alpha}^{\ell-1}(\boldsymbol{x})\right\|_2 > 0\right]$$

$$= \mathbb{E}\left[\sum_{i=1}^n \mathbb{1}_{\boldsymbol{W}_{(:,1)}^\ell>0}\,\middle|\,\left\|\boldsymbol{\alpha}^{\ell-1}(\boldsymbol{x})\right\|_2 > 0\right].$$

From the symmetry of the Gaussian distribution

$$\mathbb{E}\left[|I_\ell(\boldsymbol{x})|\,\middle|\,\left\|\boldsymbol{\alpha}^{\ell-1}(\boldsymbol{x})\right\|_2 > 0\right] = \frac{n}{2}.$$

Since this variable is a sum of $n$ independent variables taking values in $\{0,1\}$, an application of Bernstein's inequality for bounded random variables (lemma G.3) gives

$$\mathbb{P}\left[\left||I_\ell(\boldsymbol{x})| - \frac{n}{2}\right| > \frac{n}{4}\right] \leq \mathbb{P}\left[\left||I_\ell(\boldsymbol{x})| - \frac{n}{2}\right| > \frac{n}{4}\,\middle|\,\left\|\boldsymbol{\alpha}^{\ell-1}(\boldsymbol{x})\right\|_2 > 0\right] + \mathbb{P}\left[\left\|\boldsymbol{\alpha}^{\ell-1}(\boldsymbol{x})\right\|_2 = 0\right]$$

$$\leq 2\exp\left(-c'\frac{n^2/16}{n+n/4}\right) + Ce^{-cn} \leq C'e^{-c''n}$$

for appropriate constants. A union bound gives

$$\mathbb{P}\left[\bigcap_{\ell=1}^L\left\{\left||I_\ell(\boldsymbol{x})| - \frac{n}{2}\right| > \frac{n}{4}\right\}\right] \leq 2LC'e^{-c''n}$$

from which

$$\mathbb{P}\left[\min_{\ell}|I_\ell(\boldsymbol{x})| \geq \frac{n}{4}\right] \geq 1 - 2LC'e^{-c''n}$$

follows.

To prove the second inequality, we use the AM-GM inequality which gives

$$\left(\prod_{\ell=1}^{L}\frac{2\,|I_\ell(\boldsymbol{x})|}{n}\right)^{1/L} \leq \frac{2\sum_{\ell=1}^{L}|I_\ell(\boldsymbol{x})|}{nL}$$

and hence

$$\mathbb{P}\left[\prod_{\ell=1}^{L}\frac{2\,|I_\ell(\boldsymbol{x})|}{n} \geq 1+t\right] = \mathbb{P}\left[\left(\prod_{\ell=1}^{L}\frac{2\,|I_\ell(\boldsymbol{x})|}{n}\right)^{1/L} \geq (1+t)^{1/L}\right]$$

$$\leq \mathbb{P}\left[\sum_{\ell=1}^{L}\frac{2\,|I_\ell(\boldsymbol{x})|}{n} \geq L\,(1+t)^{1/L}\right]$$

Convexity of the exponential gives $(1+t)^{1/L} \geq 1+\frac{1}{L}\log(1+t)$ and for $t \leq 1$ we have $\log(1+t) \geq t\log 2$, giving

$$\mathbb{P}\left[\prod_{\ell=1}^{L}\frac{2\,|I_\ell(\boldsymbol{x})|}{n} \geq 1+t\right] \leq \mathbb{P}\left[\sum_{\ell=1}^{L}\frac{2\,|I_\ell(\boldsymbol{x})|}{n} - L \geq t\log 2\right]$$

We note that

$$\frac{2\,|I_\ell(\boldsymbol{x})|}{n} \overset{d}{=} \sum_{i=1}^{n}\mathbb{1}_{\mathcal{E}^\ell}b_i^\ell$$

where $b_i^\ell = \frac{2}{n}\theta_i^\ell, \theta_i^\ell \sim_{\text{iid}} \text{Bern}(\frac{1}{2})$ and $\mathcal{E}^\ell = \left\{\max_{i}b_i^{\ell-1} \neq 0\right\}$ is the event that the features at layer $\ell-1$ are not identically 0. Since $\sum_{i=1}^{n}\mathbb{1}_{\mathcal{E}^\ell}b_i^\ell \leq \sum_{i=1}^{n}b_i^\ell$ a.s. we have

$$\mathbb{P}\left[\sum_{\ell=1}^{L}\frac{2\,|I_\ell(\boldsymbol{x})|}{n} - L \geq t\log 2\right] \leq \mathbb{P}\left[\sum_{\ell=1}^{L}\sum_{i=1}^{n}b_i^\ell - L \geq t\log 2\right].$$

Since this is a sum of independent bounded random variables, an application of Bernstein's inequality for bounded random variables (lemma G.3) gives

$$\mathbb{P}\left[\sum_{\ell=1}^{L}\sum_{i=1}^{n}b_i^\ell - L \geq t\right] \leq 2\exp\left(-c\frac{t^2}{\sum\mathbb{E}(b_i^\ell)^2 + \frac{2}{n}t}\right) = 2\exp\left(-c'\frac{n}{L}t^2\right)$$

for some absolute constant $c'$, where we used $L \geq 1 \geq t$. Hence

$$\mathbb{P}\left[\prod_{\ell=1}^{L}\frac{2\,|I_\ell(\boldsymbol{x})|}{n} - 1 \geq t\right] \leq 2\exp\left(-c'\frac{n}{L}t^2\right).$$

$\square$

## E   SHARP BOUNDS ON THE ONE-STEP ANGLE PROCESS

In this section, we characterize the process by which angles between features for different pairs of points evolve as they are propagated across one layer of the zero-time network. This section is self-contained, and as such it will occasionally overload notation used elsewhere in the document for different local purposes. In particular, we will use the notation $\sigma(x) = [x]_+$ for the ReLU in this section (and only in this section), and $\dot{\sigma}(\boldsymbol{g}) = \mathbb{1}_{\boldsymbol{g}>0}$ for its weak derivative.

### E.1 Definitions and Preliminaries

Let $n \in \mathbb{N}$, with $n \geq 2$. Let $\boldsymbol{g}_1$ and $\boldsymbol{g}_2$ be i.i.d. $\mathcal{N}(\boldsymbol{0}, (2/n)\boldsymbol{I})$ random vectors; we use $\mu$ to denote the joint law of these random variables. We write $\boldsymbol{G} \in \mathbb{R}^{n \times 2}$ for the matrix with first column $\boldsymbol{g}_1$ and second column $\boldsymbol{g}_2$, and $\boldsymbol{g}^1, \ldots, \boldsymbol{g}^n$ for the $n$ rows of $\boldsymbol{G}$. If $S \subset [n]$ is nonempty and $\boldsymbol{A} \in \mathbb{R}^{n \times m}$, we write $\boldsymbol{A}_S \in \mathbb{R}^{|S| \times m}$ to denote the submatrix of $\boldsymbol{A}$ consisting of the rows indexed by $S$ in increasing index order. In such situations $S^{\mathrm{c}}$ will always denote the complement relative to $[n]$.

For $0 \leq \nu \leq 2\pi$, define random variables

$$\boldsymbol{v}_\nu(\boldsymbol{g}_1, \boldsymbol{g}_2) = \sigma\left(\boldsymbol{g}_1 \cos \nu + \boldsymbol{g}_2 \sin \nu\right),$$

and

$$\dot{\boldsymbol{v}}_\nu(\boldsymbol{g}_1, \boldsymbol{g}_2) = \dot{\sigma}\left(\boldsymbol{g}_1 \cos \nu + \boldsymbol{g}_2 \sin \nu\right) \odot \left(\boldsymbol{g}_2 \cos \nu - \boldsymbol{g}_1 \sin \nu\right).$$

Because $\dot{\boldsymbol{v}}_\nu$ separates over coordinates of its arguments and has each of its coordinates the product of a nondecreasing function and a continuous function, it is Borel measurable. A key property that we will use throughout this section is that the joint distribution of $(\boldsymbol{g}_1, \boldsymbol{g}_2)$ is rotationally invariant; in particular, it is invariant to rotations of the type

$$\boldsymbol{G} \mapsto \boldsymbol{G} \begin{bmatrix} \cos \nu & \sin \nu \\ \sin \nu & -\cos \nu \end{bmatrix},$$

where $\nu \in [0, 2\pi]$. Since we can write

$$\boldsymbol{v}_\nu = \sigma\left(\boldsymbol{G} \begin{bmatrix} \cos \nu \\ \sin \nu \end{bmatrix}\right), \qquad \dot{\boldsymbol{v}}_\nu = \dot{\sigma}\left(\boldsymbol{G} \begin{bmatrix} \cos \nu \\ \sin \nu \end{bmatrix}\right) \odot \left(\boldsymbol{G} \begin{bmatrix} -\sin \nu \\ \cos \nu \end{bmatrix}\right),$$

where all of the $\mathbb{R}^2$ vectors appearing above are elements of $\mathbb{S}^1$, it follows by applying rotational invariance and the specific rotation given above that

$$(\boldsymbol{v}_\nu, \dot{\boldsymbol{v}}_\nu) \stackrel{d}{=} (\boldsymbol{v}_0, -\dot{\boldsymbol{v}}_0).$$

This equivalence is useful for evaluating expectations and differentiating with respect to $\nu$.

If $0 < c \leq 0.5$ and $m \in \mathbb{N}_0$ with $m < n$, define an event

$$\mathcal{E}_{c,m} = \bigcap_{\substack{S \subset [n] \\ |S|=m}} \bigcap_{\nu \in [0, 2\pi]} \left\{ (\boldsymbol{g}_1, \boldsymbol{g}_2) \,\middle|\, c \leq \|\boldsymbol{I}_{S^{\mathrm{c}}} \boldsymbol{v}_\nu(\boldsymbol{g}_1, \boldsymbol{g}_2)\|_2 \leq c^{-1} \right\}.$$

For each $c, m$, the set $\mathcal{E}_{c,m}$ is closed, since $\|\boldsymbol{A} \boldsymbol{v}_\nu\|$ is a continuous function of $(\boldsymbol{g}_1, \boldsymbol{g}_2) \in \mathcal{E}_m$ for any linear map $\boldsymbol{A}$. We further define

$$\mathcal{E}_{0,m} = \bigcup_{k \in \mathbb{N}} \mathcal{E}_{1/(2k),m},$$

so that

$$\mathcal{E}_{0,m} = \bigcap_{\substack{S \subset [n] \\ |S|=m}} \bigcap_{\nu \in [0, 2\pi]} \left\{ (\boldsymbol{g}_1, \boldsymbol{g}_2) \,|\, 0 < \|\boldsymbol{I}_{S^{\mathrm{c}}} \boldsymbol{v}_\nu(\boldsymbol{g}_1, \boldsymbol{g}_2)\|_2 \right\},$$

and $\mathcal{E}_{0,m}$ is Borel measurable. If $c$ is omitted, we take the constant $c$ in the definition to be $0.5$. On $\mathcal{E}_{c,m}$ we guarantee that $\|\boldsymbol{v}_\nu\|_0 \geq m$ uniformly on $[0, \pi]$. Define a function $X_\nu$ by

$$X_\nu = \mathbb{1}_{\mathcal{E}_1} \left\langle \frac{\boldsymbol{v}_0}{\|\boldsymbol{v}_0\|_2}, \frac{\boldsymbol{v}_\nu}{\|\boldsymbol{v}_\nu\|_2} \right\rangle.$$

On $\mathcal{E}_1$, we guarantee that $\boldsymbol{v}_\nu \neq \boldsymbol{0}$ for every $\nu$, so $X_\nu$ is well defined; because $\mathcal{E}_1$ is Borel measurable, we have that $X_\nu$ is Borel measurable, and moreover $|X_\nu| \leq 1$, so $X_\nu \in L_\mu^p$ for every $p \geq 1$. Finally, define for $0 \leq \nu \leq \pi$

$$\bar{\varphi}(\nu) = \mathop{\mathbb{E}}_{\boldsymbol{g}_1, \boldsymbol{g}_2} \left[ \cos^{-1} X_\nu \right], \qquad \varphi(\nu) = \cos^{-1} \mathop{\mathbb{E}}_{\boldsymbol{g}_1, \boldsymbol{g}_2} \left[ \langle \boldsymbol{v}_0, \boldsymbol{v}_\nu \rangle \right].$$

E.2 MAIN RESULTS

**Lemma E.1.** *There exist absolute constants $c, C, C' > 0$ and absolute constants $K, K' > 0$ such that if $d \geq K$ and $n \geq K'd^4 \log^4 n$, then one has*

$$|\bar{\varphi}(\nu) - \varphi(\nu)| \leq C\nu \frac{\log n}{n} + C'n^{-cd}$$

*Proof.* Using the triangle inequality, we can write

$$|\bar{\varphi}(\nu) - \varphi(\nu)| \leq |\bar{\varphi}(\nu) - \cos^{-1}\mathbb{E}[X_\nu]| + |\cos^{-1}\mathbb{E}[X_\nu] - \varphi(\nu)|.$$

Choose $n$ sufficiently large to satisfy the hypotheses of Lemmas E.6 and E.7; applying these lemmas to bound the first and second terms, we conclude the claimed result (after choosing $n$ larger than an absolute constant multiple of $d \log n$ so that the $n^{-cd}$ error dominates the $e^{-c'n}$ error). $\square$

**Lemma E.2.** *One has*

$$\varphi(\nu) = \cos^{-1}\left(\left(1 - \frac{\nu}{\pi}\right)\cos\nu + \frac{\sin\nu}{\pi}\right).$$

*Proof.* See (Cho & Saul, 2009). $\square$

**Lemma E.3.** *There exist absolute constants $c, C, C', C'' > 0$ and absolute constants $K, K' > 0$ such that if $d \geq K$ and $n \geq K'd^4 \log^4 n$, then one has with probability at least $1 - C''n^{-cd}$*

$$\left|\cos^{-1} X_\nu - \varphi(\nu)\right| \leq C\nu\sqrt{\frac{d \log n}{n}} + C'n^{-cd}.$$

*The constant $c$ is the same as the constant appearing in Lemma E.1.*

*Proof.* Under our hypothess, the second result in Lemma E.6 together with Lemma E.1 and the triangle inequality imply the claimed result (after worst-casing multiplicative constants). $\square$

**Lemma E.4.** *There exist absolute constants $c, C, C' > 0$ and absolute constants $K, K' > 0$ such that if $d \geq K$ and $n \geq K'd^4 \log^4 n$, then one has*

$$\mathbb{E}\left[\left(\cos^{-1} X_\nu - \varphi(\nu)\right)^2\right] \leq C\nu^2\frac{d \log n}{n} + C'n^{-cd}.$$

*The constant $c$ is the same as the constant appearing in Lemma E.1.*

*Proof.* Under our hypotheses, Lemma E.3 is applicable; we let $\mathcal{E}$ denote the event corresponding to the bound in this lemma. By boundedness of $\cos^{-1}$, nonnegativity of $X_\nu$, and $\varphi \leq \pi/2$ from Lemma E.2, we have $\|\cos^{-1} X_\nu - \varphi(\nu)\|_{L^\infty} \leq \pi$. Thus

$$\mathbb{E}\left[\left(\cos^{-1} X_\nu - \varphi(\nu)\right)^2\right] \leq \mathbb{E}\left[\mathbb{1}_{\mathcal{E}}\left(\cos^{-1} X_\nu - \varphi(\nu)\right)^2\right] + C''\pi^2 n^{-cd}$$

$$\leq \left(C\nu\sqrt{\frac{d \log n}{n}} + C'n^{-cd}\right)^2 + C''\pi^2 n^{-cd}$$

$$\leq C^2\nu^2\frac{d \log n}{n} + C'''n^{-cd},$$

as claimed. $\square$

**Lemma E.5.** *One has*

1. *$\varphi \in C^\infty(0, \pi)$, and $\dot{\varphi}$ and $\ddot{\varphi}$ extend to continuous functions on $[0, \pi]$;*

2. *$\varphi(0) = 0$ and $\varphi(\pi) = \pi/2$; $\dot{\varphi}(0) = 1$, $\ddot{\varphi}(0) = -2/(3\pi)$, and $\dddot{\varphi}(0) = -1/(3\pi^2)$; and $\dot{\varphi}(\pi) = \ddot{\varphi}(\pi) = 0$;*

3. *$\varphi$ is concave and strictly increasing on $[0, \pi]$ (strictly concave in the interior);*

4. $\ddot{\varphi} < -c < 0$ *for an absolute constant $c > 0$ on $[0, \pi/2]$;*

5. $0 < \dot{\varphi} < 1$ *and $0 > \ddot{\varphi} \geq -C$ on $(0, \pi)$ for some absolute constant $C > 0$;*

6. $\nu(1 - C_1\nu) \leq \varphi(\nu) \leq \nu(1 - c_1\nu)$ *on $[0, \pi]$ for some absolute constants $C_1, c_1 > 0$.*

*Proof.* Deferred to Appendix E.4. $\qquad\qquad\square$

### E.3 SUPPORTING RESULTS

#### E.3.1 CORE SUPPORTING RESULTS

**Lemma E.6.** *There exist constants $c, C, C', C'', C''', C'''' > 0$ and an absolute constant $K > 0$ such that for any $d \geq 1$, if $n$ and $d$ satisfy the hypotheses of Lemmas E.9 and E.10 and moreover $n \geq Kd\log n$, then one has*

$$\left| \mathop{\mathbb{E}}_{\boldsymbol{g}_1, \boldsymbol{g}_2} \left[ \cos^{-1} X_\nu \right] - \cos^{-1} \mathop{\mathbb{E}}_{\boldsymbol{g}_1, \boldsymbol{g}_2} \left[ X_\nu \right] \right| \leq C\nu \frac{\log n}{n} + C'n^{-cd},$$

*and with probability at least $1 - C''n^{-cd}$, one has*

$$\left| \cos^{-1} X_\nu - \mathbb{E}\left[ \cos^{-1} X_\nu \right] \right| \leq C'''\nu\sqrt{\frac{d\log n}{n}} + C''''n^{-cd}.$$

*Proof.* Fix $\nu \in [0, \pi]$. The function $\cos^{-1}$ is smooth on $(-\delta, 1)$ if $0 < \delta < 1$, and Taylor expansion with Lagrange remainder on this domain about the point $\mathbb{E}[X_\nu]$ (by Lemma E.23, we have $0 \leq \mathbb{E}[X_\nu] < 1$ if $\nu > 0$; we will handle $\nu = 0$ separately below) gives

$$\cos^{-1}(x) = \cos^{-1}\mathbb{E}[X_\nu] - \frac{1}{\sqrt{1 - \mathbb{E}[X_\nu]^2}}\left(x - \mathbb{E}[X_\nu]\right) - \frac{\xi}{2\left(1 - \xi^2\right)^{3/2}}\left(x - \mathbb{E}[X_\nu]\right)^2,$$

where $\xi$ lies between $x$ and $\mathbb{E}[X_\nu]$. Using the fact that $X_\nu \neq 1$ almost surely if $\nu > 0$, which is established in Lemma E.23, we plug in $x = X_\nu$ to get

$$\cos^{-1}\mathbb{E}[X_\nu] - \cos^{-1}(X_\nu) = \frac{1}{\sqrt{1 - \mathbb{E}[X_\nu]^2}}\left(x - \mathbb{E}[X_\nu]\right) + \frac{\xi(X_\nu)}{2\left(1 - \xi(X_\nu)^2\right)^{3/2}}\left(X_\nu - \mathbb{E}[X_\nu]\right)^2,$$
$$\text{(E.1)}$$

where we now express $\xi$ as a function of $X_\nu$. From Jensen's inequality it is clear

$$\mathbb{E}[\cos^{-1} X_\nu] \leq \cos^{-1}\mathbb{E}[X_\nu], \qquad\qquad\qquad \text{(E.2)}$$

so all that remains is to obtain a matching upper bound for the righthand side of (E.1). We will make use of the following facts, proved in subsequent sections: there are absolute constants $C_i > 0$, $i \in [6]$, and $c_i > 0$, $i \in [5]$, such that

1. $\mathbb{E}[X_\nu] \leq 1 - c_5\nu^2 + C_1 e^{-c_1 n}$. (Lemma E.8)

2. For each $\nu$, $\mathrm{Var}[X_\nu] \leq C_5\nu^4 \log n/n + C_2 e^{-c_2 n}$. ( Lemma E.9)

3. With probability at least $1 - C_3 n^{-c_3 d}$, one has $|X_\nu - \mathbb{E}[X_\nu]| \leq C_6\nu^2\sqrt{d\log n/n} + C_4 e^{-c_4 n}$. (Lemma E.10)

Let $\mathcal{E}$ denote the event on which property 3 holds. Combining properties 1 and 3, we obtain with probability at least $1 - C_3 n^{-c_3 d}$

$$X_\nu \leq 1 - (c_5/2)\nu^2 + C_1 e^{-c_1 n} + C_4 e^{-c_4 n},$$

provided $n$ is chosen larger than an absolute constant multiple of $d\log n$. Thus, defining

$$\nu_0 = \frac{4}{c_5}\left(C_1 e^{-c_1 n} + C_4 e^{-c_4 n}\right),$$

we obtain for $\nu \geq \nu_0$

$$\mathbb{E}[X_\nu] \leq 1 - \frac{c_5}{4}\nu^2, \qquad X_\nu \leq 1 - \frac{c_5}{4}\nu^2, \tag{E.3}$$

with the second bound holding with probability at least $1 - C_3 n^{-c_3 d}$. Considering first $0 \leq \nu \leq \nu_0$, we obtain using the triangle inequality, Lemma E.20 and property 3

$$\begin{aligned}
\left| \cos^{-1}\mathbb{E}[X_\nu] - \mathbb{E}\left[\cos^{-1}(X_\nu)\right] \right| &\leq \mathbb{E}\left[\mathbb{1}_\mathcal{E}\left|\cos^{-1}\mathbb{E}[X_\nu] - \cos^{-1}(X_\nu)\right|\right] \\
&\quad + \mathbb{E}\left[\mathbb{1}_{\mathcal{E}^c}\left|\cos^{-1}\mathbb{E}[X_\nu] - \cos^{-1}(X_\nu)\right|\right] \\
&\leq \mathbb{E}\left[\mathbb{1}_\mathcal{E}\sqrt{|X_\nu - \mathbb{E}[X_\nu]|}\right] + \mathbb{E}[\mathbb{1}_{\mathcal{E}^c}\pi/2] \\
&\leq Ce^{-cn} + C'n^{-c'd}, \tag{E.4}
\end{aligned}$$

with the final inequality following from the triangle inequality for the $\ell^2$ norm and the fact that $\nu \leq \nu_0$. Meanwhile, if $\nu \geq \nu_0$, we have by (E.3)

$$0 \leq \xi(X_\nu) \leq \max\{X_\nu, \mathbb{E}[X_\nu]\} \leq 1 - \frac{c_5}{4}\nu^2$$

with probability at least $1 - C_3 n^{-c_3 d}$. Using $1 - x^2 = (1+x)(1-x)$ and $\mathbb{E}[X_\nu] \geq 0$, $\xi(X_\nu) \geq 0$, we have under this condition on $\nu$

$$\frac{1}{\sqrt{1 - \mathbb{E}[X_\nu]^2}} \leq \frac{1}{\sqrt{1 - \mathbb{E}[X_\nu]}} \leq \frac{2}{c_5\nu} \tag{E.5}$$

and similarly

$$\frac{\xi(X_\nu)}{2\left(1 - \xi(X_\nu)^2\right)^{3/2}} \leq \frac{4}{c_5^3\nu^3}\mathbb{1}_\mathcal{E} + \frac{\pi}{2}\mathbb{1}_{\mathcal{E}^c}. \tag{E.6}$$

Applying (E.6) and taking expectations in (E.1), we obtain by property 2

$$\cos^{-1}\mathbb{E}[X_\nu] - \mathbb{E}[\cos^{-1}X_\nu] \leq C\nu\frac{\log n}{n} + C'e^{-cn} + C''n^{-c_3 d}. \tag{E.7}$$

Together, (E.2), (E.4) and (E.7) establish the first claim provided $n$ is chosen larger than an absolute constant multiple of $d \log n$.

For the second claim, we begin by using the triangle inequality to write

$$\left|\cos^{-1}X_\nu - \mathbb{E}\left[\cos^{-1}X_\nu\right]\right| \leq \left|\cos^{-1}X_\nu - \cos^{-1}\mathbb{E}[X_\nu]\right| + \left|\cos^{-1}\mathbb{E}[X_\nu] - \mathbb{E}\left[\cos^{-1}X_\nu\right]\right|,$$

and then observe that our proof of the first claim implies suitable control of the second term. For the first term, if $\nu \leq \nu_0$ we use Lemma E.20 to immediately obtain with probability at least $1 - C_3 n^{-c_3 d}$ that this term is at most $Ce^{-cn}$. For $\nu \geq \nu_0$, we apply property 3 and the bounds (E.5) and (E.6) in the expression (E.1) to obtain with probability at least $1 - C_3 n^{-c_3 d}$

$$\left|\cos^{-1}X_\nu - \cos^{-1}\mathbb{E}[X_\nu]\right| \leq C\nu\sqrt{\frac{d\log n}{n}} + C'\nu\frac{d\log n}{n},$$

which is of the claimed order when $n$ is chosen larger than an absolute constant multiple of $d \log n$. $\qquad\square$

**Lemma E.7.** *There exist absolute constants $c, C, C', C'' > 0$ such that if $n \geq C \log n$, one has*

$$\left|\varphi(\nu) - \cos^{-1}\mathop{\mathbb{E}}_{\boldsymbol{g}_1,\boldsymbol{g}_2}[X_\nu]\right| \leq C'e^{-cn} + C''\frac{\nu}{n}.$$

*Proof.* Write $f(\nu) = \cos\varphi(\nu)$, and

$$h(\nu) = \mathbb{E}[X_\nu] - f(\nu),$$

so that $h$ is the residual between the two terms whose images we are trying to tie together. We will make use of the following results:

1. The function $\cos^{-1}$ is $\frac{1}{2}$-Hölder continuous on $[0, 1]$, so that $|\cos^{-1}x - \cos^{-1}y| \leq \sqrt{|x-y|}$ if $x, y \geq 0$. (Lemma E.20)

2. For $\nu \in [0, \pi]$, we have $1 - \frac{1}{2}\nu^2 \leq f(\nu) \leq 1 - c_2\nu^2$. (Lemma E.14)

3. For all $0 \leq \nu \leq \pi$, $|h(\nu)| \leq C_1 e^{-c_1 n} + C_2 \nu^2 / n$. (Lemma E.15)

We choose $n$ large enough that the hypotheses of Lemma E.15 are satisfied. Define $\nu_0 = 2\sqrt{C_1/c_2} e^{-c_1 n/2}$. We split the analysis into two sub-intervals: $I_1 := [0, \nu_0]$, and $I_2 := [\nu_0, \pi]$. Choosing $n$ larger than an absolute constant multiple of $\log n$, we guarantee that $I_1$ and $I_2$ both have positive measure.

On $I_1$, we proceed as follows:

$$\left|\cos^{-1} f - \cos^{-1}(f+h)\right| \leq \sqrt{|h|}$$
$$\leq \sqrt{C_1 e^{-c_1 n} + C_2 \nu^2/n}$$
$$\leq \sqrt{C_1 e^{-c_1 n} + 4C_1 C_2 c_2^{-1} e^{-c_1 n}}$$
$$\leq C e^{-\frac{1}{2} c_1 n}.$$

The first inequality uses Hölder continuity of $\cos^{-1}$, the second uses our bound on the residual, the third uses the definition of $I_1$, and the fourth worst-cases the constants.

On $I_2$, we calculate

$$|f + h| \leq |f| + |h| \leq C_1 e^{-c_1 n} + C_2 \frac{\nu^2}{n} + 1 - c_2\nu^2,$$

using the triangle inequality and our bounds on $|h|$ and $f$. Using the conditions $\nu \geq \nu_0$ and choosing $n \geq 4C_2/c_2$, we can rearrange to get

$$C_1 e^{-c_1 n} + C_2 \frac{\nu^2}{n} \leq \frac{c_2\nu^2}{2},$$

which implies $|f + h| \leq 1 - c_2\nu^2/2$. By the control $f(\nu) \leq 1 - c_2\nu^2$, valid on $I_2$, we get that both $f$ and $f + h$ are bounded above by $1 - c_2\nu^2/2$ on $I_2$; moreover, because $f \geq 0$ and $f + h \geq 0$, we can apply local Lipschitz properties of $\cos^{-1}$ on $I_2$. This yields

$$\left|\cos^{-1} f - \cos^{-1}(f+h)\right| \leq \frac{|h|}{\sqrt{1 - \left(\sup_{I_2} \max\{f, f+h\}\right)^2}}$$
$$\leq \frac{C_1 e^{-c_1 n} + C_2 \nu^2/n}{\sqrt{1 - (1 - (c_2/2)\nu^2)^2}}$$
$$= \frac{C_1 e^{-c_1 n}}{\sqrt{\frac{1}{2}c_2\nu^2(2 - \frac{1}{2}c_2\nu^2)}} + \frac{C_2 \nu^2/n}{\sqrt{\frac{1}{2}c_2\nu^2(2 - \frac{1}{2}c_2\nu^2)}}$$
$$\leq C\nu^{-1} e^{-c_1 n} + C'\nu/n$$
$$\leq C e^{-\frac{1}{2}c_1 n} + C'\nu/n.$$

Above, the first inequality is the instantiation of the local Lipschitz property; the second applies our upper and lower bounds on $f$ and $f + h$ derived above, and our bound on the residual $|h|$; the fourth applies the bound $0 \leq f(\nu) \leq 1 - \frac{1}{2}c_2\nu^2$ to conclude $2 - \frac{1}{2}c_2\nu^2 \geq 1$ on $I_2$, and cancels a factor of $\nu$ in the second term; and in the last line, we apply $\nu \in I_2$ to get $\nu \geq 2\sqrt{C_1/c_2} e^{-c_1 n/2}$, which allows us to cancel the $\nu^{-1}$ factor in the first term of the previous line.

To wrap up, we can choose the largest of the constants appearing in the bounds derived for $I_1$ and $I_2$ above and then conclude, since $I_1 \cup I_2 = [0, \pi]$ under our condition on $n$. □

### E.3.2 PROVING LEMMA E.6

**Lemma E.8.** *There exist absolute constants $c, c', C, C', C'' > 0$ such that if $n \geq C$ and if $n$ is sufficiently large to satisfy the hypotheses of Lemma E.15, one has*

$$1 - C''\nu^2 - C'e^{-c'n} \leq \mathbb{E}_{g_1, g_2}[X_\nu] \leq 1 - c\nu^2 + C'e^{-c'n}.$$

*Proof.* By the triangle inequality, we have

$$|\cos\varphi(\nu)| - |\mathbb{E}[X_\nu] - \cos\varphi(\nu)| \le \mathbb{E}[X_\nu] \le |\cos\varphi(\nu)| + |\mathbb{E}[X_\nu] - \cos\varphi(\nu)|.$$

Applying Lemmas E.14 and E.15 with $m = 0$, we get

$$1 - C''\nu^2 - Ce^{-c'n} - C'\nu^2/n \le \mathbb{E}[X_\nu] \le 1 - c\nu^2 + Ce^{-c'n} + C'\nu^2/n,$$

which proves the claim if we choose $n \ge 2C'/c$. $\qquad\square$

**Lemma E.9.** *There exist absolute constants $c, C, C' > 0$ such that if $n$ satisfies the hypotheses of Lemmas E.11 and E.12, then one has for each $\nu \in [0, \pi]$*

$$\mathrm{Var}[X_\nu] \le \frac{C\nu^4 \log n}{n} + C'e^{-cn}.$$

*Proof.* We use the following elementary fact for a random variable with finite first and second moments, easily proved using $\mathrm{Var}[X_\nu] = \mathbb{E}[X_\nu^2] - \mathbb{E}[X_\nu]^2$ and Fubini's theorem: in this setting one has

$$\mathrm{Var}[X_\nu] = \mathbb{E}_{\boldsymbol{g}_1}[\mathrm{Var}[X_\nu(\boldsymbol{g}_1, \cdot)]] + \mathrm{Var}[\mathbb{E}_{\boldsymbol{g}_2}[X_\nu(\cdot, \boldsymbol{g}_2)]].$$

By Lemma E.11, there is an event $\mathcal{E}$ of probability at least $1 - Ce^{-cn}$ on which $\mathrm{Var}[X_\nu(\boldsymbol{g}_1, \cdot)] \le C'\nu^4/n + C''e^{-c'n}$. Invoking as well Lemma E.12, we obtain

$$\mathrm{Var}[X_\nu] \le \mathbb{E}_{\boldsymbol{g}_1}[(\mathbb{1}_\mathcal{E} + \mathbb{1}_{\mathcal{E}^c})\mathrm{Var}[X_\nu(\boldsymbol{g}_1, \cdot)]] + \frac{C'''\nu^4 \log n}{n} + C''''e^{-c''n}$$

$$\le \frac{C\nu^4 \log n}{n} + C'e^{-cn} + \mathbb{P}[\mathcal{E}^c]^{1/2} \mathbb{E}_{\boldsymbol{g}_1}\left[\mathrm{Var}[X_\nu(\boldsymbol{g}_1, \cdot)]^2\right]^{1/2}$$

$$\le \frac{C\nu^4 \log n}{n} + C'e^{-cn},$$

as claimed, where in the second line we applied nonnegativity of the variance and the Schwarz inequality, and in the third line we used the fact that $\|X\|_{L^2} \le \|X\|_{L^\infty}$ for any random variable $X$ in $L^\infty$. $\qquad\square$

**Lemma E.10.** *There exist absolute constants $c, c', C, C', C'' > 0$ and absolute constants $K, K' > 0$ such that for any $d \ge 1$ such that $n$ and $d$ satisfy the hypotheses of Lemmas E.11 and E.13 and $n \ge \max\{K \log n, K'd\}$, for any $\nu \in [0, \pi]$, one has*

$$\mathbb{P}\left[\left|X_\nu - \mathbb{E}_{\boldsymbol{g}_1, \boldsymbol{g}_2}[X_\nu]\right| \le C''\nu^2\sqrt{\frac{d \log n}{n}} + Ce^{-cn}\right] \ge 1 - C'n^{-c'd}.$$

*Proof.* By Lemma E.11, we have

$$\mathbb{P}\left[\left|X_\nu - \mathbb{E}_{\boldsymbol{g}_2}[X_\nu]\right| \le C''\nu^2\sqrt{\frac{d}{n}} + Ce^{-cn}\right] \ge 1 - C'e^{-c'd}.$$

Let $\psi = \psi_{0.25}$ denote the cutoff function defined in Lemma E.31, and write

$$Y_\nu(\boldsymbol{g}_1, \boldsymbol{g}_2) = \left\langle \frac{\boldsymbol{v}_0(\boldsymbol{g}_1, \boldsymbol{g}_2)}{\psi\left(\|\boldsymbol{v}_0(\boldsymbol{g}_1, \boldsymbol{g}_2)\|_2\right)}, \frac{\boldsymbol{v}_\nu(\boldsymbol{g}_1, \boldsymbol{g}_2)}{\psi\left(\|\boldsymbol{v}_\nu(\boldsymbol{g}_1, \boldsymbol{g}_2)\|_2\right)} \right\rangle.$$

By Lemma E.13, we have

$$\mathbb{P}\left[\left|\mathbb{E}_{\boldsymbol{g}_2}[Y_\nu] - \mathbb{E}_{\boldsymbol{g}_1, \boldsymbol{g}_2}[Y_\nu]\right| \le C''\nu^2\sqrt{\frac{d \log n}{n}} + Cne^{-cn}\right] \ge 1 - C'n^{-c'd}.$$

We have $X_\nu = Y_\nu$ on the event $\mathcal{E}_1$, by Lemma E.16, and we thus calculate using the triangle inequality

$$\left|\mathbb{E}_{\boldsymbol{g}_1, \boldsymbol{g}_2}[Y_\nu] - \mathbb{E}_{\boldsymbol{g}_1, \boldsymbol{g}_2}[X_\nu]\right| \le \mathbb{E}_{\boldsymbol{g}_1, \boldsymbol{g}_2}[|X_\nu - Y_\nu|] = \mathbb{E}_{\boldsymbol{g}_1, \boldsymbol{g}_2}\left[\mathbb{1}_{\mathcal{E}_1^c}Y_\nu\right] \le Cne^{-cn},$$

where the last inequality uses Hölder's inequality and the measure bound in Lemma E.16. Again using the triangle inequality, we have

$$\left| \mathbb{E}_{\boldsymbol{g}_2}[X_\nu] - \mathbb{E}_{\boldsymbol{g}_2}[Y_\nu] \right| \leq \mathbb{E}_{\boldsymbol{g}_2}[|X_\nu - Y_\nu|],$$

and so using our previous calculation and Markov's inequality, we can assert

$$\mathbb{P}\left[ \left| \mathbb{E}_{\boldsymbol{g}_2}[Y_\nu] - \mathbb{E}_{\boldsymbol{g}_2}[X_\nu] \right| \leq Cne^{-cn/2} \right] \geq 1 - e^{-cn/2}.$$

The claim then follows from the triangle inequality, a union bound, and a choice of $n$ larger than an absolute constant multiple of $\log n$ and an absolute constant multiple of $d$. $\qquad \square$

**Lemma E.11.** *There exist absolute constants* $c, c', c'', c''', C, C', C'', C''', C_4, C_5 > 0$, *and absolute constants* $K, K' > 0$ *such that for any* $d \geq 1$, *if* $n \geq \max\{Kd, K' \log n\}$, *then for every* $\nu \in [0, \pi]$ *one has with probability at least* $1 - Ce^{-cn}$

$$\mathrm{Var}[X_\nu(\boldsymbol{g}_1, \cdot)] \leq \frac{C_4 \nu^4}{n} + C'e^{-c'n},$$

*and with* $(\boldsymbol{g}_1, \boldsymbol{g}_2)$ *probability at least* $1 - C''e^{-c''d}$ *one has*

$$\left| X_\nu - \mathbb{E}_{\boldsymbol{g}_2}[X_\nu] \right| \leq C_5 \nu^2 \sqrt{\frac{d}{n}} + C'''e^{-c'''n}.$$

*Proof.* Fix $\nu \in [0, \pi]$. Let $\mathcal{E}_1 = \mathcal{E}_{0.5,1}$ denote the event in Lemma E.16 which is in the definition of $X_\nu$. We start by treating the case of $\nu = 0$ or $\nu = \pi$. We have $X_\pi = 0$ deterministically, so the variance is zero and it equals its partial expectation over $\boldsymbol{g}_2$ with probability one. For the other case, one has $X_0 = \mathbb{1}_{\mathcal{E}_1}$; we have

$$\mathrm{Var}[X_0(\boldsymbol{g}_1, \cdot)] = \mathbb{E}_{\boldsymbol{g}_2}[\mathbb{1}_{\mathcal{E}_1}] - \mathbb{E}_{\boldsymbol{g}_2}[\mathbb{1}_{\mathcal{E}_1}]^2 \leq \left( 1 - \mathbb{E}_{\boldsymbol{g}_2}[\mathbb{1}_{\mathcal{E}_1}] \right),$$

and since $\mathbb{E}[\mathbb{1}_{\mathcal{E}_1}] = 1 - Cne^{-cn}$ by Lemma E.16, we obtain by Markov's inequality

$$\mathbb{P}\left[ \mathrm{Var}[X_0(\boldsymbol{g}_1, \cdot)] \geq Cne^{-cn/2} \right] \leq e^{-cn/2}.$$

This gives a suitable bound on the variance with suitable probability. For deviations, we note that

$$\mathbb{E}\left[ X_0 - \mathbb{E}_{\boldsymbol{g}_2}[X_0] \right] = 0,$$

and following our previous variance inequality but taking expectations over both $\boldsymbol{g}_1$ and $\boldsymbol{g}_2$ gives $\mathrm{Var}[X_0] \leq \tilde{C}ne^{-cn}$, which implies by Chebyshev's inequality

$$\mathbb{P}\left[ \left| X_0 - \mathbb{E}_{\boldsymbol{g}_2}[X_0] \right| \geq \sqrt{\tilde{C}ne^{-cn/2}} \right] \leq e^{-cn/2}$$

which is a suitable deviations bound that we can union bound with the event constructed below, which controls deviations uniformly for the remaining values of $\nu$. We therefore assume below that $0 < \nu < \pi$.

Let $\psi(x) = \max\{x, \frac{1}{8}\}$, which is continuous and differentiable except at $x = \frac{1}{8}$, with derivative $\psi'(x) = \mathbb{1}_{x > 1/8}$. We note in addition that $x \leq \psi(x)$, and since $\psi \geq \frac{1}{8}$ we have for $x \geq 0$ the bound $x/\psi(x) \leq 1$. Define

$$Y_\nu(\boldsymbol{g}_1, \boldsymbol{g}_2) = \frac{\langle \boldsymbol{v}_0(\boldsymbol{g}_1, \boldsymbol{g}_2), \boldsymbol{v}_\nu(\boldsymbol{g}_1, \boldsymbol{g}_2) \rangle}{\psi(\|\boldsymbol{v}_0(\boldsymbol{g}_1, \boldsymbol{g}_2)\|_2)\psi(\|\boldsymbol{v}_\nu(\boldsymbol{g}_1, \boldsymbol{g}_2)\|_2)}.$$

We first show that it is enough to prove the claims for $Y_\nu$, which will be preferable for technical reasons. On $\mathcal{E}_1$, we have $Y_\nu = X_\nu$. We have $|Y_\nu| \leq 1$, and we calculate

$$\mathbb{E}_{\boldsymbol{g}_2}\left[(Y_\nu - X_\nu)^2\right] = \mathbb{E}_{\boldsymbol{g}_2}\left[\mathbb{1}_{\mathcal{E}_1^c}(Y_\nu - X_\nu)^2\right] \leq \mathbb{E}_{\boldsymbol{g}_2}\left[\mathbb{1}_{\mathcal{E}_1^c}\right]^{1/2} \mathbb{E}_{\boldsymbol{g}_2}\left[(Y_\nu - X_\nu)^4\right]^{1/2} \leq C \mathbb{E}_{\boldsymbol{g}_2}\left[\mathbb{1}_{\mathcal{E}_1^c}\right]^{1/2},$$

where the first inequality uses the Schwarz inequality, and the last inequality uses that $|X_\nu| \leq 1$ and the triangle inequality, and where $C > 0$ is an absolute constant. We have by Tonelli's theorem and Lemma E.16

$$\mathop{\mathbb{E}}_{g_1}\left[\mathop{\mathbb{E}}_{g_2}\left[\mathbb{1}_{\mathcal{E}_1^c}\right]^{1/2}\right] \leq Cne^{-cn},$$

so Markov's inequality implies

$$\mathbb{P}\left[\mathop{\mathbb{E}}_{g_2}\left[\mathbb{1}_{\mathcal{E}_1^c}\right]^{1/2} \geq Cne^{-cn/2}\right] \leq e^{-cn/2}.$$

Thus, with probability at least $1 - e^{-cn/2}$, we have

$$\mathop{\mathbb{E}}_{g_2}\left[(Y_\nu - X_\nu)^2\right] \leq C'ne^{-cn/2},$$

so that an application of Lemma E.32 yields that with probability at least $1 - e^{-cn/2}$

$$\mathrm{Var}[X_\nu(g_1, \cdot)] \leq \mathrm{Var}[Y_\nu(g_1, \cdot)] + C''ne^{-c'n},$$

where we have worst-cased constants and the exponent on $n$. For deviations, we write using the triangle inequality

$$\left|X_\nu - \mathop{\mathbb{E}}_{g_2}[X_\nu]\right| \leq |X_\nu - Y_\nu| + \left|Y_\nu - \mathop{\mathbb{E}}_{g_2}[Y_\nu]\right| + \left|\mathop{\mathbb{E}}_{g_2}[Y_\nu] - \mathop{\mathbb{E}}_{g_2}[X_\nu]\right|,$$

and then note that the first term is identically zero on the event $\mathcal{E}_1$, which has probability at least $1 - Ce^{-cn}$, whereas for the third term, we have

$$\left|\mathop{\mathbb{E}}_{g_2}[Y_\nu] - \mathop{\mathbb{E}}_{g_2}[X_\nu]\right| \leq \mathop{\mathbb{E}}_{g_2}\left[(Y_\nu - X_\nu)^2\right]^{1/2} \leq C'ne^{-cn/2},$$

where the first inequality uses the triangle inequality and the Lyapunov inequality, and the second inequality holds with probability at least $1 - e^{-cn/2}$, and leverages the argument we used to control the difference in variances. Ultimately taking union bounds, we can conclude that it sufficient to prove the claimed properties for $Y_\nu$.

With $0 < \nu < \pi$ fixed, we introduce the notation

$$u_{g_1} = v_0(g_1); \quad v_{g_1,g_2} = v_\nu(g_1, g_2),$$

so that

$$Y_\nu = \left\langle \frac{u_{g_1}}{\psi(\|u_{g_1}\|_2)}, \frac{v_{g_1,g_2}}{\psi(\|v_{g_1,g_2}\|_2)} \right\rangle.$$

For fixed $g_1$, we will write $Y_\nu(g_2) = Y_\nu(g_1, g_2)$ with an abuse of notation. For $\bar{g} \in \mathbb{R}^n$ arbitrary and $g_2$ fixed, we consider the function $f(t) = Y_\nu(g_2 + t\bar{g})$ for $t \in [0, 1]$. Writing $f'$ for the derivative of $f$ where it exists, at any point of differentiability, we calculate by the chain rule

$$f'(t) = \langle \nabla_{g_2} Y_\nu(g_2 + t\bar{g}), \bar{g} \rangle,$$

where

$$\nabla_{g_2} Y_\nu(g_2) = \frac{\sin\nu}{\psi(\|u_{g_1}\|_2)\psi(\|v_{g_1,g_2}\|_2)} \left(I - \frac{\psi'(\|v_{g_1,g_2}\|_2)v_{g_1,g_2}v_{g_1,g_2}^*}{\psi(\|v_{g_1,g_2}\|_2)\|v_{g_1,g_2}\|_2}\right) \left(\mathbb{1}_{v_{g_1,g_2}>0} \odot u_{g_1}\right).$$

Using the fact that

$$\mathbb{1}_{v_{g_1,g_2}>0} \odot u_{g_1} = P_{\{v_{g_1,g_2}>0\}} u_{g_1},$$

where $P_{\{v_{g_1,g_2}>0\}}$ is the orthogonal projection onto the coordinates where $v_{g_1,g_2}$ is positive, together with the fact that

$$v_{g_1,g_2}v_{g_1,g_2}^* P_{\{v_{g_1,g_2}>0\}} = P_{\{v_{g_1,g_2}>0\}} v_{g_1,g_2}v_{g_1,g_2}^*,$$

we can also write

$$\nabla_{g_2} Y_\nu(g_2) = \frac{\sin\nu}{\psi(\|u_{g_1}\|_2)\psi(\|v_{g_1,g_2}\|_2)} P_{\{v_{g_1,g_2}>0\}} \left(I - \frac{\psi'(\|v_{g_1,g_2}\|_2)v_{g_1,g_2}v_{g_1,g_2}^*}{\psi(\|v_{g_1,g_2}\|_2)\|v_{g_1,g_2}\|_2}\right) u_{g_1}.$$

$$\tag{E.8}$$

We next argue that $f$ does not fail to be differentiable at too many points of $[0, 1]$. Because $\psi > 0$, it will suffice to show that (i) $t \mapsto \boldsymbol{v}_{\boldsymbol{g}_1, \boldsymbol{g}_2 + t\bar{\boldsymbol{g}}}$ and (ii) $t \mapsto \psi(\|\boldsymbol{v}_{\boldsymbol{g}_1, \boldsymbol{g}_2 + t\bar{\boldsymbol{g}}}\|_2)$ are differentiable at all but a set of isolated points in $[0, 1]$. For the latter function, we note that at any point where $\|\boldsymbol{v}_{\boldsymbol{g}_1, \boldsymbol{g}_2 + t\bar{\boldsymbol{g}}}\|_2 < \frac{1}{8}$, by continuity we have that $t \mapsto \psi(\|\boldsymbol{v}_{\boldsymbol{g}_1, \boldsymbol{g}_2 + t\bar{\boldsymbol{g}}}\|_2)$ is locally constant, and therefore differentiable at such points. At other points, by Lemma E.21 it suffices to characterize $t \mapsto \|\boldsymbol{v}_{\boldsymbol{g}_1, \boldsymbol{g}_2 + t\bar{\boldsymbol{g}}}\|_2$ as differentiable at all but isolated points, and because $\|\boldsymbol{v}_{\boldsymbol{g}_1, \boldsymbol{g}_2 + t\bar{\boldsymbol{g}}}\|_2 \geq \frac{1}{8}$ by assumption, the norm is differentiable and by the chain rule it suffices to characterize differentiability of each coordinate of $t \mapsto \boldsymbol{v}_{\boldsymbol{g}_1, \boldsymbol{g}_2 + t\bar{\boldsymbol{g}}}$, which settles the question of all-but-isolated differentiability of (i) as well. We have $\boldsymbol{v}_{\boldsymbol{g}_1, \boldsymbol{g}_2 + t\bar{\boldsymbol{g}}} = \sigma(\boldsymbol{g}_1 \cos \nu + \boldsymbol{g}_2 \sin \nu + t\bar{\boldsymbol{g}} \sin \nu)$, so again by Lemma E.21, we conclude from differentiability of $t \mapsto \boldsymbol{g}_1 \cos \nu + \boldsymbol{g}_2 \sin \nu + t\bar{\boldsymbol{g}} \sin \nu$ that $t \mapsto \boldsymbol{v}_{\boldsymbol{g}_1, \boldsymbol{g}_2 + t\bar{\boldsymbol{g}}}$ is differentiable at all but isolated points, and consequently so is $f$. In particular, $f$ is differentiable at all but countably many points of $[0, 1]$. Next, we show that $f'$ has suitable integrability properties. Indeed, we calculate using (E.8)

$$
\begin{aligned}
\|\nabla_{\boldsymbol{g}_2} Y_\nu(\boldsymbol{g}_2)\|_2 &\leq 8\nu \left\| \left( \boldsymbol{I} - \frac{\psi'(\|\boldsymbol{v}_{\boldsymbol{g}_1, \boldsymbol{g}_2}\|_2) \boldsymbol{v}_{\boldsymbol{g}_1, \boldsymbol{g}_2} \boldsymbol{v}_{\boldsymbol{g}_1, \boldsymbol{g}_2}^*}{\psi(\|\boldsymbol{v}_{\boldsymbol{g}_1, \boldsymbol{g}_2}\|_2) \|\boldsymbol{v}_{\boldsymbol{g}_1, \boldsymbol{g}_2}\|_2} \right) \frac{\boldsymbol{u}_{\boldsymbol{g}_1}}{\psi(\|\boldsymbol{u}_{\boldsymbol{g}_1}\|_2)} \right\|_2 \\
&= 8\nu \sqrt{1 - \psi'(\|\boldsymbol{v}_{\boldsymbol{g}_1, \boldsymbol{g}_2}\|_2) Y_\nu(\boldsymbol{g}_2)^2},
\end{aligned}
\tag{E.9}
$$

where we used Cauchy-Schwarz and $\psi \geq \frac{1}{8}$ in the first inequality and distributed and applied $(\psi')^2 = \psi'$ and the estimate $x/\psi(x) \leq 1$ in the second inequality. In particular, this implies that $|f'(t)| \leq C\|\bar{\boldsymbol{g}}\|_2$, which is a $t$-integrable upper bound for every $\bar{\boldsymbol{g}}$. Because $Y_\nu(\boldsymbol{g}_1, \cdot)$ is continuous by continuity of $\sigma$, $\psi$, and the fact that $\psi$ becomes constant whenever $\|\boldsymbol{v}_{\boldsymbol{g}_1, \boldsymbol{g}_2}\|_2 < \frac{1}{8}$, we can apply (Cohn, 2013, Theorem 6.3.11) to get

$$
Y_\nu(\boldsymbol{g}_2 + \bar{\boldsymbol{g}}) = Y_\nu(\boldsymbol{g}_2) + \int_0^1 \langle \nabla_{\boldsymbol{g}_2} Y_\nu(\boldsymbol{g}_2 + t\bar{\boldsymbol{g}}), \bar{\boldsymbol{g}} \rangle \, \mathrm{d}t,
$$

and since $\bar{\boldsymbol{g}}$ was arbitrary, for any $\boldsymbol{g}_2' \in \mathbb{R}^n$ we can put $\bar{\boldsymbol{g}} = \boldsymbol{g}_2' - \boldsymbol{g}_2$ to get

$$
Y_\nu(\boldsymbol{g}_2') = Y_\nu(\boldsymbol{g}_2) + \int_0^1 \langle \nabla_{\boldsymbol{g}_2} Y_\nu(t\boldsymbol{g}_2' + (1 - t)\boldsymbol{g}_2), \boldsymbol{g}_2' - \boldsymbol{g}_2 \rangle \, \mathrm{d}t.
$$

Performing the expansion with $\boldsymbol{g}_2$ and $\boldsymbol{g}_2'$ reversed and applying the triangle inequality and Cauchy-Schwarz then implies the estimate

$$
|Y_\nu(\boldsymbol{g}_2') - Y_\nu(\boldsymbol{g}_2)| \leq \|\boldsymbol{g}_2' - \boldsymbol{g}_2\|_2 \int_0^1 \|\nabla_{\boldsymbol{g}_2} Y_\nu(t\boldsymbol{g}_2' + (1 - t)\boldsymbol{g}_2)\|_2 \, \mathrm{d}t.
\tag{E.10}
$$

This relation is enough to conclude the result for angles satisfying $\nu \geq c_0$, where $0 < c_0 \leq \pi/4$ is an absolute constant. Indeed, (E.9) and (E.10) imply that $Y_\nu$ is $C$-Lipschitz, where $C > 0$ is an absolute constant; so the Gaussian Poincaré inequality implies

$$
\mathbb{E}_{\boldsymbol{g}_2} \left[ \left( Y_\nu - \mathbb{E}_{\boldsymbol{g}_2}[Y_\nu] \right) \right] \leq \frac{C'}{n},
$$

and Gauss-Lipschitz concentration implies for any $d \geq 0$

$$
\mathbb{P} \left[ \left| Y_\nu - \mathbb{E}_{\boldsymbol{g}_2}[Y_\nu] \right| \geq C'' \sqrt{\frac{d}{n}} \right] \leq 2e^{-d}.
$$

Because $\nu \geq c_0$, we can adjust these bounds to involve $\nu^4$ and $\nu^2$ (respectively) while only paying increases in the constant factors. We proceed assuming $0 < \nu \leq c_0$.

Let $0 \leq \tau_{\boldsymbol{g}_1} \leq 1$ denote a median of $Y_\nu(\boldsymbol{g}_1, \cdot)$, i.e., a number satisfying $\mathbb{P}_{\boldsymbol{g}_2}[Y_\nu \geq \tau_{\boldsymbol{g}_1}] \geq \frac{1}{2}$ and $\mathbb{P}_{\boldsymbol{g}_2}[Y_\nu \leq \tau_{\boldsymbol{g}_1}] \geq \frac{1}{2}$, and for each $0 \leq s < \tau_{\boldsymbol{g}_1}$ define

$$
w_s(\boldsymbol{g}_2) = \max\{Y_\nu(\boldsymbol{g}_2), \tau_{\boldsymbol{g}_1} - s\}.
$$

For any $0 \leq s < \tau_{\boldsymbol{g}_1}$, notice that $w_s \geq Y_\nu$, which implies that $\mathbb{P}[w_s \geq \tau_{\boldsymbol{g}_1}] \geq \mathbb{P}[Y_\nu \geq \tau_{\boldsymbol{g}_1}] \geq \frac{1}{2}$, because $\tau_{\boldsymbol{g}_1}$ is a median of $Y_\nu$; and similarly $\mathbb{P}[w_s \leq \tau_{\boldsymbol{g}_1}] \geq \mathbb{P}[Y_\nu \leq \tau_{\boldsymbol{g}_1}] \geq \frac{1}{2}$, so that $\tau_{\boldsymbol{g}_1}$ is also a median of $w_s$. The fact that $w_s \geq Y_\nu$ implies for any $t > 0$ that $\mathbb{P}[Y_\nu - \tau_{\boldsymbol{g}_1} > t] \leq \mathbb{P}[w_s - \tau_{\boldsymbol{g}_1} > t]$,

and additionally if $Y_\nu \le \tau_{\boldsymbol{g}_1} - s$ we have $w_s = \tau_{\boldsymbol{g}_1} - s$, so that $\mathbb{P}[Y_\nu - \tau_{\boldsymbol{g}_1} \le -s] \le \mathbb{P}[w_s - \tau_{\boldsymbol{g}_1} \le -s]$. In particular, the tails of $Y_\nu$ can be controlled in terms of those of $w_s$ for appropriate choices of $s$. Additionally, by Lemma E.21, we have that for each $s$, $t \mapsto w_s(\boldsymbol{g}_2 + t\bar{\boldsymbol{g}})$ is differentiable at all but countably many points of $[0, 1]$, and has derivative there equal to $t \mapsto \langle \bar{\boldsymbol{g}}, \nabla_{\boldsymbol{g}_2} w_s(\boldsymbol{g}_2) \rangle$, where

$$\nabla_{\boldsymbol{g}_2} w_s(\boldsymbol{g}_2) = \mathbb{1}_{w_s(\boldsymbol{g}_2) > \tau_{\boldsymbol{g}_1} - s} \nabla_{\boldsymbol{g}_2} Y_\nu(\boldsymbol{g}_2),$$

which, following from (E.9), satisfies a strengthened gradient norm estimate

$$\|\nabla_{\boldsymbol{g}_2} w_s(\boldsymbol{g}_2)\|_2 \le 8\nu \mathbb{1}_{w_s(\boldsymbol{g}_2) > \tau_{\boldsymbol{g}_1} - s} \sqrt{1 - \psi'(\|\boldsymbol{v}_{\boldsymbol{g}_1, \boldsymbol{g}_2}\|_2) Y_\nu(\boldsymbol{g}_2)^2}$$
$$\le 8\nu \sqrt{1 - \psi'(\|\boldsymbol{v}_{\boldsymbol{g}_1, \boldsymbol{g}_2}\|_2)(\tau_{\boldsymbol{g}_1} - s)^2}. \tag{E.11}$$

In particular, we obtain a nearly-Lipschitz estimate of the form (E.10):

$$|w_s(\boldsymbol{g}_2') - w_s(\boldsymbol{g}_2)| \le \|\boldsymbol{g}_2' - \boldsymbol{g}_2\|_2 \int_0^1 8\nu \sqrt{1 - \psi'(\|\boldsymbol{v}_{\boldsymbol{g}_1, t\boldsymbol{g}_2' + (1-t)\boldsymbol{g}_2}\|_2)(\tau_{\boldsymbol{g}_1} - s)^2} \, \mathrm{d}t. \tag{E.12}$$

For each $\boldsymbol{g}_1$, we define a set $S_{\boldsymbol{g}_1} = \{\boldsymbol{g}_2 \mid \|\boldsymbol{v}_{\boldsymbol{g}_1, \boldsymbol{g}_2}\|_2 \ge \frac{1}{4}\}$. Noting that the function $\boldsymbol{g}_2 \mapsto \|\sigma(\boldsymbol{g}_1 \cos \nu + \boldsymbol{g}_2 \sin \nu)\|_2$ is a convex 1-Lipschitz function (given that $|\sin \nu| \le 1$), we have by Gauss-Lipschitz concentration

$$\mathbb{P}_{\boldsymbol{g}_2}\left[\|\boldsymbol{v}_{\boldsymbol{g}_1, \boldsymbol{g}_2}\|_2 \le \mathbb{E}_{\boldsymbol{g}_2}[\|\boldsymbol{v}_{\boldsymbol{g}_1, \boldsymbol{g}_2}\|_2] - t\right] \le e^{-cnt^2},$$

and by Jensen's inequality

$$\mathbb{E}_{\boldsymbol{g}_2}[\|\boldsymbol{v}_{\boldsymbol{g}_1, \boldsymbol{g}_2}\|_2] \ge |\cos \nu| \|\boldsymbol{u}_{\boldsymbol{g}_1}\|_2 \ge \frac{\|\boldsymbol{u}_{\boldsymbol{g}_1}\|_2}{\sqrt{2}},$$

where the last line holds because $\nu \le \pi/4$. By Lemma E.16, there is a $\boldsymbol{g}_1$ event $\mathcal{E}$ having probability at least $1 - Ce^{-cn}$ on which $\|\boldsymbol{u}_{\boldsymbol{g}_1}\|_2 \ge \frac{1}{2}$, so that for any $\boldsymbol{g}_1 \in \mathcal{E}$, we have by a suitable choice of $t$ in our Gauss-Lipschitz bound $\mathbb{P}_{\boldsymbol{g}_2}[S_{\boldsymbol{g}_1}] \ge 1 - e^{-cn}$. Thus, using the first line of (E.11), the Gaussian Poincaré inequality and the Lipschitz property of $w_s$ (which follows from (E.12) after bounding by an absolute constant) and Rademacher's theorem on a.e. differentiability of Lipschitz functions, we have whenever $\boldsymbol{g}_1 \in \mathcal{E}$

$$\begin{aligned}
\mathrm{Var}[w_s] &\le \frac{2}{n} \mathbb{E}_{\boldsymbol{g}_2}\left[\|\nabla_{\boldsymbol{g}_2} w_s(\boldsymbol{g}_2)\|_2^2\right] \le \frac{128\nu^2}{n} \mathbb{E}_{\boldsymbol{g}_2}\left[\left(1 - \psi'(\|\boldsymbol{v}_{\boldsymbol{g}_1, \boldsymbol{g}_2}\|_2) Y_\nu(\boldsymbol{g}_2)^2\right)\right] \\
&\le \frac{128\nu^2}{n} \mathbb{E}_{\boldsymbol{g}_2}\left[\mathbb{1}_{\mathcal{E}}\left(1 - \psi'(\|\boldsymbol{v}_{\boldsymbol{g}_1, \boldsymbol{g}_2}\|_2) Y_\nu(\boldsymbol{g}_2)^2\right)\right] \\
&\qquad + \mathbb{E}_{\boldsymbol{g}_2}\left[\mathbb{1}_{\mathcal{E}^c}\left(1 - \psi'(\|\boldsymbol{v}_{\boldsymbol{g}_1, \boldsymbol{g}_2}\|_2) Y_\nu(\boldsymbol{g}_2)^2\right)\right] \\
&\le \frac{256\nu^2}{n} \mathbb{E}_{\boldsymbol{g}_2}[1 - Y_\nu(\boldsymbol{g}_2)] + Ce^{-cn}, \tag{E.13}
\end{aligned}$$

where we also make use of the fact that $0 \le Y_\nu \le 1$. Now, we calculate for $\boldsymbol{g}_1 \in \mathcal{E}$ and $\boldsymbol{g}_2 \in S_{\boldsymbol{g}_1}$

$$\begin{aligned}
Y_\nu &= 1 - \frac{1}{2} \left\| \frac{\boldsymbol{u}_{\boldsymbol{g}_1}}{\|\boldsymbol{u}_{\boldsymbol{g}_1}\|_2} - \frac{\boldsymbol{v}_{\boldsymbol{g}_1, \boldsymbol{g}_2}}{\|\boldsymbol{v}_{\boldsymbol{g}_1, \boldsymbol{g}_2}\|_2} \right\|_2^2 \\
&\ge 1 - 2\frac{\|\boldsymbol{u}_{\boldsymbol{g}_1} - \boldsymbol{v}_{\boldsymbol{g}_1, \boldsymbol{g}_2}\|_2^2}{\|\boldsymbol{u}_{\boldsymbol{g}_1}\|_2^2} \\
&\ge 1 - 8\|\boldsymbol{u}_{\boldsymbol{g}_1} - \boldsymbol{v}_{\boldsymbol{g}_1, \boldsymbol{g}_2}\|_2^2 \\
&\ge 1 - 8\|\boldsymbol{g}_1 - (\boldsymbol{g}_1 \cos \nu + \boldsymbol{g}_2 \sin \nu)\|_2^2, \tag{E.14}
\end{aligned}$$

where the second inequality uses $\boldsymbol{g}_1 \in \mathcal{E}$, the third uses nonexpansiveness of $\sigma$, and the first requires a proof; we will show that for any nonzero vectors $\boldsymbol{x}, \boldsymbol{y} \in \mathbb{R}^n$, one has

$$\left\| \frac{\boldsymbol{x}}{\|\boldsymbol{x}\|_2} - \frac{\boldsymbol{y}}{\|\boldsymbol{y}\|_2} \right\|_2 \le 2\frac{\|\boldsymbol{x} - \boldsymbol{y}\|_2}{\|\boldsymbol{y}\|_2}. \tag{E.15}$$

To see this, write $\theta$ for the angle between $\boldsymbol{x}$ and $\boldsymbol{y}$, and distribute to obtain equivalently

$$-\frac{1}{2}\|\boldsymbol{y}\|_2^2(1 + \cos\theta) \le \|\boldsymbol{x}\|_2^2 - 2\|\boldsymbol{x}\|_2\|\boldsymbol{y}\|_2\cos\theta.$$

Divide through by $\|\boldsymbol{x}\|_2^2$, write $K = \|\boldsymbol{y}\|_2\|\boldsymbol{x}\|_2^{-1}$, and rearrange to obtain the equivalent expression

$$K^2(1 + \cos\theta)) - 4K\cos\theta + 2 \ge 0.$$

It suffices to minimize the LHS of the previous inequality with respect to $K$ subject to the constraint $K > 0$ and then study the resulting function of $\theta$ to ascertain the validity of the bound. Given that $1 + \cos\theta \ge 0$, the LHS is a convex function of $K$, with minimizer $K = 2\cos\theta(1 + \cos\theta)^{-1}$, and therefore for any $\theta \ge \pi/2$, the LHS subject to the constraint $K > 0$ is minimized at $K = 0$, where the inequality is easily seen to be true. If $\theta < \pi/2$, we have that the minimizer is positive, and we verify that after substituting the bound becomes

$$1 + \cos\theta \ge 2\cos^2\theta,$$

which is also seen to be true for $\theta < \pi/2$, for example by showing that the polynomial $x \mapsto -2x^2 + x + 1$ is nonnegative on $[0, 1]$. This proves the inequality, so returning to (E.14), we have

$$\begin{aligned} Y_\nu &\ge 1 - 8((1 - \cos\nu)^2\|\boldsymbol{g}_1\|_2^2 + \sin^2\nu\|\boldsymbol{g}_2\|_2^2 - 2(\sin\nu)(1 - \cos\nu)\langle\boldsymbol{g}_1, \boldsymbol{g}_2\rangle) \\ &\ge 1 - 8((1 - \cos\nu)^2\|\boldsymbol{g}_1\|_2^2 + \sin^2\nu\|\boldsymbol{g}_2\|_2^2 - 2(\sin\nu)(1 - \cos\nu)\|\boldsymbol{g}_1\|_2\|\boldsymbol{g}_2\|_2) \end{aligned}$$

using Cauchy-Schwarz in the second inequality. By Gauss-Lipschitz concentration (e.g. following the proof of the third assertion in Lemma E.17), there is a $\boldsymbol{g}_1$ event $\mathcal{E}'$ and a $\boldsymbol{g}_2$ event $\mathcal{E}''$, each with probability at least $1 - Ce^{-cn}$, on which we have (respectively) $\|\boldsymbol{g}_i\|_2 \le 2$ for $i = 1, 2$. Then using $(\sin\nu)(1 - \cos\nu) \ge 0$, we obtain that when $\boldsymbol{g}_1 \in \mathcal{E} \cap \mathcal{E}'$ and when $\boldsymbol{g}_2 \in S_{\boldsymbol{g}_1} \cap \mathcal{E}''$

$$Y_\nu \ge 1 - 32((1 - \cos\nu)^2 + \sin^2\nu) = 1 - 64(1 - \cos\nu) \ge 1 - 32\nu^2,$$

where the final inequality uses the standard estimate $\cos\nu \ge 1 - 0.5\nu^2$, which can be proved via Taylor expansion. By a union bound, we can assert that with $\boldsymbol{g}_1$-probability at least $1 - Ce^{-cn}$, with conditional (in $\boldsymbol{g}_2$) probability at least $1 - C'e^{-c'n}$ we have $Y_\nu \ge 1 - 32\nu^2$, so that in particular, by nonnegativity of $Y_\nu$, and choosing $n$ larger than an absolute constant, we guarantee with $\boldsymbol{g}_1$-probability at least $1 - Ce^{-cn}$

$$\mathbb{E}_{\boldsymbol{g}_2}[Y_\nu] \ge 1 - 32\nu^2 - C'e^{-cn}, \qquad \tau_{\boldsymbol{g}_1} \ge 1 - 32\nu^2. \tag{E.16}$$

Plugging the mean estimate into (E.13), we conclude with probability at least $1 - C''e^{-c'n}$

$$\mathrm{Var}[w_s] \le \frac{C\nu^4}{n} + C'e^{-cn}. \tag{E.17}$$

We could have just as well applied this exact argument to $Y_\nu$ instead of $w_s$, so we conclude the claimed variance bound from this expression. We have stated the result in terms of the truncations $w_s$ so that it can be applied towards deviations control in the sequel. As an immediate application, we use the fact that any median is a minimizer of the quantity $c \mapsto \mathbb{E}[|X - c|]$ for any integrable $X$ and $c \in \mathbb{R}$ to get with probability at least $1 - C''e^{-c'n}$

$$\left|\mathbb{E}_{\boldsymbol{g}_2}[w_s] - \tau_{\boldsymbol{g}_1}\right| \le \mathbb{E}_{\boldsymbol{g}_2}[|w_s - \tau_{\boldsymbol{g}_1}|] \le \mathbb{E}_{\boldsymbol{g}_2}\left[\left|w_s - \mathbb{E}_{\boldsymbol{g}_2}[w_s]\right|\right] \le \sqrt{\mathrm{Var}[w_s]} \le \frac{C\nu^2}{\sqrt{n}} + C'e^{-cn}, \quad \text{(E.18)}$$

where we also applied Jensen's inequality for the first inequality and the Lyapunov inequality for the third. In particular, the same argument yields

$$\left|\mathbb{E}_{\boldsymbol{g}_2}[Y_\nu] - \tau_{\boldsymbol{g}_1}\right| \le \frac{C\nu^2}{\sqrt{n}} + C'e^{-cn}. \tag{E.19}$$

We turn to removing the $t$ dependence in (E.12) without sacrificing the dependence on $\tau_{\boldsymbol{g}_1}$. To obtain a Lipschitz estimate on the subset $S_{\boldsymbol{g}_1}$ we need to control the norm of $\nabla_{\boldsymbol{g}_2}w_s$ on the line segment between $\boldsymbol{g}_2, \boldsymbol{g}_2' \in S_{\boldsymbol{g}_1}$. For this, write $\sigma_y(x) = \max\{x - y, 0\}$ for any $y \in \mathbb{R}$, and make the following observations:

1. $\boldsymbol{v}_{\boldsymbol{g}_1,\boldsymbol{g}_2} = (\sin\nu)\sigma_{-\boldsymbol{g}_1\cot\nu}(\boldsymbol{g}_2)$, so that

$$Y_\nu(\boldsymbol{g}_2) = \left\langle \frac{\boldsymbol{u}_{\boldsymbol{g}_1}}{\psi\left(\|\boldsymbol{u}_{\boldsymbol{g}_1}\|_2\right)}, \frac{(\sin\nu)\sigma_{-\boldsymbol{g}_1\cot\nu}(\boldsymbol{g}_2)}{\psi\left(\|(\sin\nu)\sigma_{-\boldsymbol{g}_1\cot\nu}(\boldsymbol{g}_2)\|_2\right)} \right\rangle;$$

2. for any $x, y$, $\sigma_y(x) = \max\{x, y\} - y$; $\boldsymbol{x} \mapsto \max\{\boldsymbol{x}, \boldsymbol{y}\}$ is the projection onto the convex set $\{\boldsymbol{x} \mid x_i \geq y_i \,\forall i\}$, so in particular $\boldsymbol{x} \mapsto \sigma_{\boldsymbol{y}}(\boldsymbol{x})$ is nonexpansive, has convex range, and satisfies $\sigma_{\boldsymbol{y}}(\sigma_{\boldsymbol{y}}(\boldsymbol{x}) + \boldsymbol{y}) = \sigma_{\boldsymbol{y}}(\boldsymbol{x})$; and thus

3. for any $\boldsymbol{g}_2$, $Y_\nu(\boldsymbol{g}_2) = Y_\nu(\sigma_{-\boldsymbol{g}_1\cot\nu}(\boldsymbol{g}_2) - \boldsymbol{g}_1\cot\nu)$.

We write $\tilde{\boldsymbol{g}}_2 = \sigma_{-\boldsymbol{g}_1\cot\nu}(\boldsymbol{g}_2) - \boldsymbol{g}_1\cot\nu$, $\tilde{\boldsymbol{g}}_2' = \sigma_{-\boldsymbol{g}_1\cot\nu}(\boldsymbol{g}_2') - \boldsymbol{g}_1\cot\nu$, so that (E.12) becomes

$$|w_s(\boldsymbol{g}_2') - w_s(\boldsymbol{g}_2)| = |w_s(\tilde{\boldsymbol{g}}_2') - w_s(\tilde{\boldsymbol{g}}_2)|$$

$$\leq \|\tilde{\boldsymbol{g}}_2' - \tilde{\boldsymbol{g}}_2\|_2 \int_0^1 8\nu\sqrt{1 - \psi'(\|\boldsymbol{v}_{\boldsymbol{g}_1,t\tilde{\boldsymbol{g}}_2'+(1-t)\tilde{\boldsymbol{g}}_2}\|_2)(\tau_{\boldsymbol{g}_1} - s)^2}\, \mathrm{d}t$$

$$\leq \|\boldsymbol{g}_2' - \boldsymbol{g}_2\|_2 \int_0^1 8\nu\sqrt{1 - \psi'(\|\boldsymbol{v}_{\boldsymbol{g}_1,t\tilde{\boldsymbol{g}}_2'+(1-t)\tilde{\boldsymbol{g}}_2}\|_2)(\tau_{\boldsymbol{g}_1} - s)^2}\, \mathrm{d}t,$$

where the second line follows from nonexpansiveness and translation invariance of the distance. Having reduced to the study of points along the segment between $\tilde{\boldsymbol{g}}_2$ and $\tilde{\boldsymbol{g}}_2'$, we now observe

$$\sigma_{-\boldsymbol{g}_1\cot\nu}(t\tilde{\boldsymbol{g}}_2' + (1-t)\tilde{\boldsymbol{g}}_2) = \sigma(t\sigma_{-\boldsymbol{g}_1\cot\nu}(\boldsymbol{g}_2) + (1-t)\sigma_{-\boldsymbol{g}_1\cot\nu}(\boldsymbol{g}_2))$$

$$= t\sigma_{-\boldsymbol{g}_1\cot\nu}(\boldsymbol{g}_2) + (1-t)\sigma_{-\boldsymbol{g}_1\cot\nu}(\boldsymbol{g}_2),$$

because $\sigma_{-\boldsymbol{g}_1\cot\nu}$ has image included in the nonnegative orthant, which is convex. It then follows from (1) above that

$$\|\boldsymbol{v}_{\boldsymbol{g}_1,t\tilde{\boldsymbol{g}}_2'+(1-t)\tilde{\boldsymbol{g}}_2}\|_2 = (\sin\nu)\|t\sigma_{-\boldsymbol{g}_1\cot\nu}(\boldsymbol{g}_2) + (1-t)\sigma_{-\boldsymbol{g}_1\cot\nu}(\boldsymbol{g}_2)\|_2$$

$$= \left\|t\boldsymbol{v}_{\boldsymbol{g}_1,\boldsymbol{g}_2} + (1-t)\boldsymbol{v}_{\boldsymbol{g}_1,\boldsymbol{g}_2'}\right\|_2,$$

and in particular

$$\left\|t\boldsymbol{v}_{\boldsymbol{g}_1,\boldsymbol{g}_2} + (1-t)\boldsymbol{v}_{\boldsymbol{g}_1,\boldsymbol{g}_2'}\right\|_2^2 = t^2\|\boldsymbol{v}_{\boldsymbol{g}_1,\boldsymbol{g}_2}\|_2^2 + 2t(1-t)\langle\boldsymbol{v}_{\boldsymbol{g}_1,\boldsymbol{g}_2}, \boldsymbol{v}_{\boldsymbol{g}_1,\boldsymbol{g}_2'}\rangle + (1-t)^2\|\boldsymbol{v}_{\boldsymbol{g}_1,\boldsymbol{g}_2'}\|_2^2$$

$$\geq \frac{1}{16}\left(t^2 + (1-t)^2\right) \geq \frac{1}{32},$$

where the first inequality uses that $\sigma \geq 0$ and $\boldsymbol{g}_2, \boldsymbol{g}_2' \in S_{\boldsymbol{g}_1}$, and the second minimizes the convex function of $t$ in the previous bound. We conclude that $\boldsymbol{g}_2, \boldsymbol{g}_2' \in S_{\boldsymbol{g}_1}$ implies that $\|\boldsymbol{v}_{\boldsymbol{g}_1,t\tilde{\boldsymbol{g}}_2'+(1-t)\tilde{\boldsymbol{g}}_2}\|_2 > \frac{1}{8}$ for every $t \in [0,1]$, and consequently (E.12) becomes (after an additional simplification of the quantity under the square root using $\tau_{\boldsymbol{g}_1} \leq 1$)

$$|w_s(\boldsymbol{g}_2') - w_s(\boldsymbol{g}_2)| \leq 16\nu\sqrt{1 - (\tau_{\boldsymbol{g}_1} - s)}\|\boldsymbol{g}_2' - \boldsymbol{g}_2\|_2, \tag{E.20}$$

so that $w_s$ is $16\nu\sqrt{1 - (\tau_{\boldsymbol{g}_1} - s)}$-Lipschitz on $S_{\boldsymbol{g}_1}$. Then by an application of the median bound in (E.16), if $0 \leq s < 1 - 32\nu^2$, with $\boldsymbol{g}_1$ probability at least $1 - Ce^{-cn}$ we have that $w_s$ is $16\nu\sqrt{32\nu^2 + s}$-Lipschitz on $S_{\boldsymbol{g}_1}$. For the previous assertion to be nonvacuous, we need to take $\nu$ small; in particular, we have $1 - 32\nu^2 > \frac{1}{2}$ if $\nu < 1/8$, which we can take to be the value of the absolute constant $c_0$ we left unspecified previously. Then for each such $s$, define $L_s = 16\nu\sqrt{32\nu^2 + s}$, and define

$$\hat{w}_s(\boldsymbol{g}_2) = \sup_{\boldsymbol{g}_2' \in S_{\boldsymbol{g}_1}} \{w_s(\boldsymbol{g}_2') - L_s\|\boldsymbol{g}_2' - \boldsymbol{g}_2\|_2\}.$$

Then $\hat{w}_s$ is $L_s$-Lipschitz on $\mathbb{R}^n$, and satisfies $\hat{w}_s = w_s$ on $S_{\boldsymbol{g}_1}$ (Evans & Gariepy, 1991, §3.1.1 Theorem 1). By the Gaussian Poincaré inequality, we obtain immediately $\mathrm{Var}[\hat{w}_s] \leq L_s$, and using $\hat{w}_s = w_s$ on $S_{\boldsymbol{g}_1}$, we compute

$$\left|\mathbb{E}_{\boldsymbol{g}_2}[w_s - \hat{w}_s]\right| = \left|\mathbb{E}_{\boldsymbol{g}_2 \in S_{\boldsymbol{g}_1}^c}[w_s - \hat{w}_s]\right| \leq \mathbb{E}_{\boldsymbol{g}_2}\left[\mathbb{1}_{S_{\boldsymbol{g}_1}^c}|w_s - \hat{w}_s|\right]$$

$$\leq \mathbb{P}_{\boldsymbol{g}_2}\left[S_{\boldsymbol{g}_1}^c\right]^{1/2}\|w_s - \hat{w}_s\|_{L^2}$$

$$\leq Ce^{-cn}\left(\|w_s\|_{L^2} + \|\hat{w}_s\|_{L^2}\right) \leq C'e^{-cn}, \tag{E.21}$$

where the second inequality follows from the Schwarz inequality, the third holds given that $\boldsymbol{g}_1 \in \mathcal{E}$ and by the Minkowski inequality, and the final uses that $w_s$ and $\hat{w}_s$ are both Lipschitz with Lipschitz constants bounded above by absolute constants together with the Gaussian Poincaré inequality. Meanwhile, by Gauss-Lipschitz concentration, we obtain a Bernstein-type lower tail

$$\mathbb{P}_{\boldsymbol{g}_2}\left[\hat{w}_s \leq \mathbb{E}_{\boldsymbol{g}_2}[\hat{w}_s] - s\right] \leq \exp\left(-\frac{cns^2}{\nu^2(32\nu^2 + s)}\right), \tag{E.22}$$

and for the upper tail, it will be sufficient to consider $\hat{w}_0$, which satisfies a subgaussian tail (for any $t \geq 0$)

$$\mathbb{P}_{\boldsymbol{g}_2}\left[\hat{w}_0 \leq \mathbb{E}_{\boldsymbol{g}_2}[\hat{w}_0] - t\right] \leq \exp\left(-\frac{c'nt^2}{\nu^4}\right). \tag{E.23}$$

Using the results (E.18), (E.19), (E.21), and the fact that $w_s = \hat{w}_s$ on $S_{\boldsymbol{g}_1}$, we get

$$\mathbb{P}_{\boldsymbol{g}_2}\left[Y_\nu - \mathbb{E}_{\boldsymbol{g}_2}[Y_\nu] \leq -s\right] \leq \mathbb{P}_{\boldsymbol{g}_2}\left[\hat{w}_s - \mathbb{E}_{\boldsymbol{g}_2}[\hat{w}_s] \leq C\frac{\nu^2}{\sqrt{n}} + C'e^{-cn} - s\right] + \mathbb{P}_{\boldsymbol{g}_2}\left[S_{\boldsymbol{g}_1}^{\mathsf{c}}\right]. \tag{E.24}$$

Using $d \geq 1$, we put $s = 2C\nu^2\sqrt{d/n} + C'e^{-cn}$ in this bound; using that $\nu < 1/8$, and in particular $1 - 32\nu^2 > \frac{1}{2}$, we can choose $n$ larger than an absolute constant multiple of $d$ to guarantee that for all $0 \leq \nu < 1/8$, this choice of $s$ is less than $1 - 32\nu^2$, and that $C\nu^2\sqrt{d/n} \leq 32\nu^2$. Together with the lower tail bound (E.22), these facts imply

$$\mathbb{P}_{\boldsymbol{g}_2}\left[Y_\nu - \mathbb{E}_{\boldsymbol{g}_2}[Y_\nu] \leq -2C\nu^2\sqrt{\frac{d}{n}} - C'e^{-cn}\right] \leq \mathbb{P}_{\boldsymbol{g}_2}\left[\hat{w}_s - \mathbb{E}_{\boldsymbol{g}_2}[\hat{w}_s] \leq -C\nu^2\sqrt{\frac{d}{n}}\right] + C''e^{-c'n}$$

$$\leq e^{-c''d} + C''e^{-c'n}.$$

Meanwhile, for the upper tail, we have for any $t \geq 0$

$$\mathbb{P}_{\boldsymbol{g}_2}\left[Y_\nu - \mathbb{E}_{\boldsymbol{g}_2}[Y_\nu] \geq t\right] \leq \mathbb{P}_{\boldsymbol{g}_2}\left[\hat{w}_0 - \mathbb{E}_{\boldsymbol{g}_2}[\hat{w}_0] \geq t - C\frac{\nu^2}{\sqrt{n}} - C'e^{-cn}\right] + \mathbb{P}_{\boldsymbol{g}_2}\left[S_{\boldsymbol{g}_1}^{\mathsf{c}}\right], \tag{E.25}$$

and if we put $t = 2C\nu^2\sqrt{d/n} + C'e^{cn}$, our previous requirements on $n$ and the upper tail bound (E.23) yield

$$\mathbb{P}_{\boldsymbol{g}_2}\left[Y_\nu - \mathbb{E}_{\boldsymbol{g}_2}[Y_\nu] \geq 2C\nu^2\sqrt{\frac{d}{n}} + C'e^{-cn}\right] \leq e^{-c'''d} + C''e^{-c'n}.$$

Combining these two bounds gives control of absolute deviations about the mean. By independence, we conclude

$$\mathbb{P}_{\boldsymbol{g}_1,\boldsymbol{g}_2}\left[\left|Y_\nu - \mathbb{E}_{\boldsymbol{g}_2}[Y_\nu]\right| \leq 2C\nu^2\sqrt{\frac{d}{n}} + C'e^{-c'''n}\right] \geq (1 - 2e^{-cd} - Ce^{-c'n})(1 - C'e^{-c''n})$$

$$\geq 1 - 2e^{-cd} - Ce^{-c'n} - C'e^{-c''n}.$$

To conclude, we have shown that for every $\nu \in [0, \pi]$ one has with probability at least $1 - Ce^{-cn}$

$$\mathrm{Var}[X_\nu(\boldsymbol{g}_1, \cdot)] \leq \frac{C'\nu^4}{n} + C''ne^{-c'n},$$

and with $(\boldsymbol{g}_1, \boldsymbol{g}_2)$ probability at least $1 - 2e^{-c''d} + C'''ne^{-c'''n}$ one has

$$\left|X_\nu - \mathbb{E}_{\boldsymbol{g}_2}[X_\nu]\right| \leq C''''\nu^2\sqrt{\frac{d}{n}} + C'''''ne^{-c''''n}.$$

To simplify these bounds, we may in addition choose $n$ larger than an absolute constant multiple of $\log n$, and $n$ larger than an absolute constant multiple of $d$, to obtain that with probability at least $1 - Ce^{-cn}$

$$\mathrm{Var}[X_\nu(\boldsymbol{g}_1, \cdot)] \leq \frac{C_4\nu^4}{n} + C'e^{-c'n},$$

and with $(\boldsymbol{g}_1, \boldsymbol{g}_2)$ probability at least $1 - C''e^{-c''d}$ one has

$$\left|X_\nu - \mathbb{E}_{\boldsymbol{g}_2}[X_\nu]\right| \leq C_5\nu^2\sqrt{\frac{d}{n}} + C'''e^{-c'''n},$$

which was to be shown. $\qquad\square$

**Lemma E.12.** *There exist absolute constants $c, C, C' > 0$ and an absolute constant $K > 0$ such that if $n \geq K \log^4 n$, then for every $\nu \in [0, \pi]$ one has*

$$\mathrm{Var}\left[\mathbb{E}_{\boldsymbol{g}_2}[X_\nu(\,\cdot\,, \boldsymbol{g}_2)]\right] \leq \frac{C\nu^4 \log n}{n} + C'e^{-cn}.$$

*Proof.* Define

$$Y_\nu(\boldsymbol{g}_1, \boldsymbol{g}_2) = \frac{\langle \boldsymbol{v}_0(\boldsymbol{g}_1, \boldsymbol{g}_2), \boldsymbol{v}_\nu(\boldsymbol{g}_1, \boldsymbol{g}_2) \rangle}{\psi(\|\boldsymbol{v}_0(\boldsymbol{g}_1, \boldsymbol{g}_2)\|_2)\psi(\|\boldsymbol{v}_\nu(\boldsymbol{g}_1, \boldsymbol{g}_2)\|_2)},$$

where $\psi = \psi_{0.25}$ is as in Lemma E.31. Then by Cauchy-Schwarz and property 2 in Lemma E.31 (the case where either $\|\boldsymbol{v}_0\|_2 = 0$ or $\|\boldsymbol{v}_\nu\|_2 = 0$ is treated separately, since in this case $Y_\nu = 0$), we obtain $|Y_\nu| \leq 4$, and

$$\mathbb{E}_{\boldsymbol{g}_1}\left[\left(\mathbb{E}_{\boldsymbol{g}_2}[X_\nu] - \mathbb{E}_{\boldsymbol{g}_2}[Y_\nu]\right)^2\right] = \mathbb{E}_{\boldsymbol{g}_1}\left[\left(\mathbb{E}_{\boldsymbol{g}_2}\left[\mathbb{1}_{\mathcal{E}^c}\frac{\langle \boldsymbol{v}_0, \boldsymbol{v}_\nu \rangle}{\psi(\|\boldsymbol{v}_0\|_2)\psi(\|\boldsymbol{v}_\nu\|_2)}\right]\right)^2\right]$$

$$\leq \mathbb{E}_{\boldsymbol{g}_1, \boldsymbol{g}_2}\left[\left(\mathbb{1}_{\mathcal{E}^c}\frac{\langle \boldsymbol{v}_0, \boldsymbol{v}_\nu \rangle}{\psi(\|\boldsymbol{v}_0\|_2)\psi(\|\boldsymbol{v}_\nu\|_2)}\right)^2\right]$$

$$\leq 16\mu(\mathcal{E}_m^c) \leq Cne^{-cn},$$

where we use the fact that if $(\boldsymbol{g}_1, \boldsymbol{g}_2) \in \mathcal{E}_m$ then $\|\boldsymbol{v}_\nu\|_2 \geq \frac{1}{2}$ for every $0 \leq \nu \leq \pi$ and hence $\psi(\|\boldsymbol{v}_\nu\|_2) = \|\boldsymbol{v}_\nu\|_2$ in the first line, apply Jensen's inequality in the second line, and combine our bound on $Y_\nu$ with Hölder's inequality and the measure bound in Lemma E.16 in the third line. An application of Lemma E.32 then yields

$$\mathrm{Var}\left[\mathbb{E}_{\boldsymbol{g}_2}[X_\nu(\,\cdot\,, \boldsymbol{g}_2)]\right] \leq \mathrm{Var}\left[\mathbb{E}_{\boldsymbol{g}_2}[Y_\nu(\,\cdot\,, \boldsymbol{g}_2)]\right] + Cne^{-cn} \leq \mathrm{Var}\left[\mathbb{E}_{\boldsymbol{g}_2}[Y_\nu(\,\cdot\,, \boldsymbol{g}_2)]\right] + Ce^{-cn/2},$$

where the last inequality holds when $n$ is chosen to be larger than an absolute constant multiple of $\log n$. It thus it suffices to control the variance of $Y_\nu$. Applying Lemma E.26, we get for almost all $\boldsymbol{g}_1 \in \mathbb{R}^n$

$$\mathbb{E}_{\boldsymbol{g}_2}[Y_\nu(\boldsymbol{g}_1, \boldsymbol{g}_2)] = \frac{\|\boldsymbol{v}_0\|_2^2}{\psi(\|\boldsymbol{v}_0\|_2)^2} + \int_0^\nu \int_0^t \mathbb{E}_{\boldsymbol{g}_2}[(\Xi_1 + \Xi_2 + \Xi_3 + \Xi_4 + \Xi_5 + \Xi_6)(s, \boldsymbol{g}_1, \boldsymbol{g}_2)]\, \mathrm{d}s\, \mathrm{d}t,$$

where we follow the notation defined in Lemma E.13. We start by removing the term outside of the integral from consideration. We have as above $|Y_\nu| \leq 4$, so that $|\mathbb{E}_{\boldsymbol{g}_2}[Y_\nu]| \leq 4$. Moreover, following the proof of the measure bound in Lemma E.16, but using only the pointwise concentration result, we assert that if $n \geq C$ an absolute constant there is an event $\mathcal{E}$ on which $0.5 \leq \|\boldsymbol{v}_0\|_2 \leq 2$ with probability at least $1 - 2e^{-cn}$ with $c > 0$ an absolute constant. This implies that if $\boldsymbol{g}_1 \in \mathcal{E}$ we have

$$\frac{\|\boldsymbol{v}_0\|_2^2}{\psi(\|\boldsymbol{v}_0\|_2)^2} = 1,$$

and since

$$\left|\frac{\|\boldsymbol{v}_0\|_2^2}{\psi(\|\boldsymbol{v}_0\|_2)^2}\right| \leq 4,$$

by the same argument used for $Y_\nu$, we can calculate

$$\left\|\frac{\|\boldsymbol{v}_0\|_2^2}{\psi(\|\boldsymbol{v}_0\|_2)^2} - 1\right\|_{L^2} \leq \left\|\left(\frac{\|\boldsymbol{v}_0\|_2^2}{\psi(\|\boldsymbol{v}_0\|_2)^2} - 1\right)\mathbb{1}_{\mathcal{E}^c}\right\|_{L^2} \leq 5\|\mathbb{1}_{\mathcal{E}^c}\|_{L^2} \leq Ce^{-cn},$$

by the Minkowski inequality and the triangle inequality. An application of Lemma E.32 implies that it is therefore sufficient to control the variance of the quantity

$$f(\nu, \boldsymbol{g}_1) = 1 + \int_0^\nu \int_0^t \mathbb{E}_{\boldsymbol{g}_2}[(\Xi_1 + \Xi_2 + \Xi_3 + \Xi_4 + \Xi_5 + \Xi_6)(s, \boldsymbol{g}_1, \boldsymbol{g}_2)]\, \mathrm{d}s\, \mathrm{d}t.$$

By Lemma E.37, the Lyapunov inequality, and Fubini's theorem, we have

$$(f(\nu, \boldsymbol{g}_1) - \mathbb{E}[f(\nu, \boldsymbol{g}_1)])^2 = \left(\int_0^\nu \int_0^t \left(\sum_{i=1}^6 \mathbb{E}_{\boldsymbol{g}_2}[\Xi_i(s, \boldsymbol{g}_1, \boldsymbol{g}_2)] - \mathbb{E}_{\boldsymbol{g}_1, \boldsymbol{g}_2}[\Xi_i(s, \boldsymbol{g}_1, \boldsymbol{g}_2)]\right) \mathrm{d}s\, \mathrm{d}t\right)^2.$$

Using the elementary inequality

$$\left(\int_0^\nu \int_0^t g(s)\,\mathrm{d}s\,\mathrm{d}t\right)^2 \le \nu \int_0^\nu t\,\mathrm{d}t \int_0^t g^2(s)\,\mathrm{d}s,$$

valid for any square integrable $g : [0, \pi] \to \mathbb{R}$ and proved with two applications of Jensen's inequality, and Lemma E.37, we obtain

$$(f(\nu, \boldsymbol{g}_1) - \mathbb{E}[f(\nu, \boldsymbol{g}_1)])^2 \le \nu \int_0^\nu t \int_0^t \left(\sum_{i=1}^6 \mathbb{E}_{\boldsymbol{g}_2}[\Xi_i(s, \boldsymbol{g}_1, \boldsymbol{g}_2)] - \mathbb{E}_{\boldsymbol{g}_1, \boldsymbol{g}_2}[\Xi_i(s, \boldsymbol{g}_1, \boldsymbol{g}_2)]\right)^2 \mathrm{d}s\,\mathrm{d}t.$$

Thus, again by Lemma E.37, the Lyapunov inequality, Fubini's theorem, and compactness of $[0, \pi]$, we have

$$\mathrm{Var}[f(\nu, \cdot)] \le \nu \int_0^\nu t \int_0^t \mathrm{Var}\left[\sum_{i=1}^6 \mathbb{E}_{\boldsymbol{g}_2}[\Xi_i(s, \boldsymbol{g}_1, \boldsymbol{g}_2)]\right] \mathrm{d}s\,\mathrm{d}t. \tag{E.26}$$

We can control the variance under the integral using a combination of Lemmas E.35 and E.37, together with the deviations control given by Lemmas E.39, E.41 to E.44 and E.46, since we have chosen $n$ according to the hypotheses of Lemma E.13. In particular, these lemmas furnish deviation bounds of size at most

$$C_i\left(\sqrt{\frac{d \log n}{n}} + n^{-c_i d}\right) + C_i' n e^{-c_i' n}$$

that hold with probabilities at least $1 - C_i'' n^{-c_i'' d} - C_i''' n e^{-c_i''' n}$, for any $d \ge 1$ larger than an absolute constant and suitable absolute constants specified above. We can simplify these bounds as follows: first, choose $n$ such that $n \ge (2/c_i''') \log n$ for each $i$, which guarantees that the bounds hold with probability at least $1 - C_i'' n^{-c_i'' d} - C_i''' e^{-c_i''' n/2}$. Next, choose $n \ge (2c_i''/c_i''') d \log n$ for all $i$, which implies that the bounds hold with probability at least $1 - 2\max\{C_i'', C_i'''\} n^{-c_i'' d}$. Similarly, we also choose $n$ such that $n \ge (2/c_i') \log n$ for each $i$, which guarantees that the error terms that are exponential in $n$ in the bounds are upper bounded by $C_i' e^{-c_i' n/2}$, and, choose $n \ge (2c_i/c_i') d \log n$ for all $i$, which implies that for all $i$

$$C_i\left(\sqrt{\frac{d \log n}{n}} + n^{-c_i d}\right) + C_i' n e^{-c_i' n} \le C_i \sqrt{\frac{d \log n}{n}} + 2\max\{C_i, C_i'\} n^{-c_i d}.$$

Finally, we make the particular choice $d = 4/\min_i\{c_i, c_i''\}$, or the minimum required value of $d$, whichever is larger, so that there are absolute constants $C, C', C'' > 0$ such that with probability at least $1 - C'' n^{-4}$ we have for all $i$

$$\left|\mathbb{E}_{\boldsymbol{g}_2}[\Xi_i(\nu, \boldsymbol{g}_1, \boldsymbol{g}_2)] - \mathbb{E}_{\boldsymbol{g}_1, \boldsymbol{g}_2}[\Xi_i(\nu, \boldsymbol{g}_1, \boldsymbol{g}_2)]\right| \le C\sqrt{\frac{\log n}{n}} + C' n^{-4} \le 2C\sqrt{\frac{\log n}{n}},$$

where the last inequality holds when $n$ is larger than an absolute constant. With these bounds, we can now invoke Lemma E.35 with Lemma E.37 to get

$$\mathrm{Var}\left[\sum_{i=1}^6 \mathbb{E}_{\boldsymbol{g}_2}[\Xi_i(s, \boldsymbol{g}_1, \boldsymbol{g}_2)]\right] \le C\frac{\log n}{n} + \frac{C'}{n^2} \le C''\frac{\log n}{n},$$

for different absolute constants $C, C', C'' > 0$, and where the last inequality again holds $n$ is larger than an absolute constant. Plugging back into (E.26) and evaluating the integrals, we get

$$\mathrm{Var}[f(\nu, \cdot)] \le C\nu^4 \frac{\log n}{n},$$

which is enough to conclude. $\qquad\qquad\square$

**Lemma E.13.** *Write*

$$Y_\nu(\boldsymbol{g}_1, \boldsymbol{g}_2) = \frac{\langle \boldsymbol{v}_0(\boldsymbol{g}_1, \boldsymbol{g}_2), \boldsymbol{v}_\nu(\boldsymbol{g}_1, \boldsymbol{g}_2)\rangle}{\psi(\|\boldsymbol{v}_0(\boldsymbol{g}_1, \boldsymbol{g}_2)\|_2)\psi(\|\boldsymbol{v}_\nu(\boldsymbol{g}_1, \boldsymbol{g}_2)\|_2)},$$

*where $\psi = \psi_{0.25}$ is as in Lemma E.31. There exist absolute constants $c, c', C, C', C'' > 0$ and absolute constants $K, K' > 0$ such that for any $d \ge 1$, if $n \ge Kd^4 \log^4 n$ and if $d \ge K'$, then there is an event $\mathcal{E}$ such that*

1. *One has*

$$\forall \nu \in [0, \pi], \quad \left| \mathbb{E}_{\boldsymbol{g}_2}[Y_\nu] - \mathbb{E}_{\boldsymbol{g}_1, \boldsymbol{g}_2}[Y_\nu] \right| \leq C'' \nu^2 \sqrt{\frac{d \log n}{n}} + C e^{-cn}$$

*if $\boldsymbol{g}_1 \in \mathcal{E}$;*

2. *One has*

$$\mathbb{P}[\mathcal{E}] \geq 1 - C' n^{-c'd}.$$

*Proof.* Fix $d > 0$, and write

$$f(\nu, \boldsymbol{g}_1) = \mathbb{E}_{\boldsymbol{g}_2}[Y_\nu(\boldsymbol{g}_1, \boldsymbol{g}_2)].$$

Applying Lemma E.26, we get for almost all $\boldsymbol{g}_1 \in \mathbb{R}^n$

$$f(\nu, \boldsymbol{g}_1) = \frac{\|\boldsymbol{v}_0\|_2^2}{\psi(\|\boldsymbol{v}_0\|_2)^2} + \int_0^\nu \int_0^t \mathbb{E}_{\boldsymbol{g}_2}[(\Xi_1 + \Xi_2 + \Xi_3 + \Xi_4 + \Xi_5 + \Xi_6)(s, \boldsymbol{g}_1, \boldsymbol{g}_2)] \, \mathrm{d}s \, \mathrm{d}t, \quad \text{(E.27)}$$

where

$$\Xi_1(s, \boldsymbol{g}_1, \boldsymbol{g}_2) = \sum_{i=1}^n \frac{\sigma(g_{1i})^3 \rho(-g_{1i} \cot s)}{\psi(\|\boldsymbol{v}_0\|_2) \psi(\|\boldsymbol{v}_s^i\|_2) \sin^3 s}$$

$$\Xi_2(s, \boldsymbol{g}_1, \boldsymbol{g}_2) = \frac{\langle \boldsymbol{v}_0, \boldsymbol{v}_s \rangle \psi'(\|\boldsymbol{v}_s\|_2) \|\boldsymbol{v}_s\|_2}{\psi(\|\boldsymbol{v}_0\|_2) \psi(\|\boldsymbol{v}_s\|_2)^2} - \frac{\langle \boldsymbol{v}_0, \boldsymbol{v}_s \rangle}{\psi(\|\boldsymbol{v}_0\|_2) \psi(\|\boldsymbol{v}_s\|_2)}$$

$$\Xi_3(s, \boldsymbol{g}_1, \boldsymbol{g}_2) = -\frac{\langle \boldsymbol{v}_0, \boldsymbol{v}_s \rangle \langle \boldsymbol{v}_s, \dot{\boldsymbol{v}}_s \rangle^2 \psi''(\|\boldsymbol{v}_s\|_2)}{\psi(\|\boldsymbol{v}_0\|_2) \psi(\|\boldsymbol{v}_s\|_2)^2 \|\boldsymbol{v}_s\|_2^2}$$

$$\Xi_4(s, \boldsymbol{g}_1, \boldsymbol{g}_2) = -2 \frac{\langle \boldsymbol{v}_0, \dot{\boldsymbol{v}}_s \rangle \langle \boldsymbol{v}_s, \dot{\boldsymbol{v}}_s \rangle \psi'(\|\boldsymbol{v}_s\|_2)}{\psi(\|\boldsymbol{v}_0\|_2) \psi(\|\boldsymbol{v}_s\|_2)^2 \|\boldsymbol{v}_s\|_2}$$

$$\Xi_5(s, \boldsymbol{g}_1, \boldsymbol{g}_2) = -\frac{\langle \boldsymbol{v}_0, \boldsymbol{v}_s \rangle \|\dot{\boldsymbol{v}}_s\|_2^2 \psi'(\|\boldsymbol{v}_s\|_2)}{\psi(\|\boldsymbol{v}_0\|_2) \psi(\|\boldsymbol{v}_s\|_2)^2 \|\boldsymbol{v}_s\|_2}$$

$$\Xi_6(s, \boldsymbol{g}_1, \boldsymbol{g}_2) = 2 \frac{\langle \boldsymbol{v}_0, \boldsymbol{v}_s \rangle \langle \boldsymbol{v}_s, \dot{\boldsymbol{v}}_s \rangle^2 \psi'(\|\boldsymbol{v}_s\|_2)}{\psi(\|\boldsymbol{v}_0\|_2) \psi(\|\boldsymbol{v}_s\|_2)^3 \|\boldsymbol{v}_s\|_2^2} + \frac{\langle \boldsymbol{v}_0, \boldsymbol{v}_s \rangle \langle \boldsymbol{v}_s, \dot{\boldsymbol{v}}_s \rangle^2 \psi'(\|\boldsymbol{v}_s\|_2)}{\psi(\|\boldsymbol{v}_0\|_2) \psi(\|\boldsymbol{v}_s\|_2)^2 \|\boldsymbol{v}_s\|_2^3}.$$

Here we put $\Xi_1(0, \boldsymbol{g}_1, \boldsymbol{g}_2) = \Xi_1(\pi, \boldsymbol{g}_1, \boldsymbol{g}_2) = 0$, which does not affect the integral and which is equal to the limits $\lim_{\nu \searrow 0} \Xi_1(\nu, \boldsymbol{g}_1, \boldsymbol{g}_2) = \lim_{\nu \nearrow \pi} \Xi_1(\nu, \boldsymbol{g}_1, \boldsymbol{g}_2)$ for every $(\boldsymbol{g}_1, \boldsymbol{g}_2)$.

Momentarily ignoring measurability issues, it is of interest to construct $\boldsymbol{g}_1$ events $\mathcal{E}_i$ of suitable probability on which we have

$$\sup_{\nu \in [0, \pi]} \left| \mathbb{E}_{\boldsymbol{g}_2}[\Xi_i(\nu, \boldsymbol{g}_1, \boldsymbol{g}_2)] - \mathbb{E}_{\boldsymbol{g}_1, \boldsymbol{g}_2}[\Xi_i(\nu, \boldsymbol{g}_1, \boldsymbol{g}_2)] \right| \leq C_i \left( \sqrt{\frac{d \log n}{n}} + n^{-c_i d} \right) + C_i' n e^{-c_i' n} \quad \text{(E.28)}$$

for each $i = 1, \ldots, 6$, and a $\boldsymbol{g}_1$ event $\mathcal{E}_7$ on which we have

$$\left| \frac{\|\boldsymbol{v}_0\|_2^2}{\psi(\|\boldsymbol{v}_0\|_2)^2} - \mathbb{E}_{\boldsymbol{g}_1}\left[ \frac{\|\boldsymbol{v}_0\|_2^2}{\psi(\|\boldsymbol{v}_0\|_2)^2} \right] \right| \leq C_7' e^{-c_7' n}.$$

We can then consider the event $\mathcal{E} = \bigcap_{i=1}^7 \mathcal{E}_i$, possibly minus a negligible set on which (E.27) fails to hold, which has high probability via a union bound and on which we have simultaneously for all $\nu \in [0, \pi]$

$$\left| f(\nu, \boldsymbol{g}_1) - \mathbb{E}_{\boldsymbol{g}_1}[f(\nu, \boldsymbol{g}_1)] \right| \leq \left| \frac{\|\boldsymbol{v}_0\|_2^2}{\psi(\|\boldsymbol{v}_0\|_2)^2} - \mathbb{E}_{\boldsymbol{g}_1}\left[ \frac{\|\boldsymbol{v}_0\|_2^2}{\psi(\|\boldsymbol{v}_0\|_2)^2} \right] \right|$$

$$+ \sum_{i=1}^6 \int_0^\nu \int_0^t \left| \mathbb{E}_{\boldsymbol{g}_2}[\Xi_i(s, \boldsymbol{g}_1, \boldsymbol{g}_2)] - \mathbb{E}_{\boldsymbol{g}_1, \boldsymbol{g}_2}[\Xi_i(s, \boldsymbol{g}_1, \boldsymbol{g}_2)] \right| \mathrm{d}s \, \mathrm{d}t$$

$$\leq C \nu^2 \left( \sqrt{\frac{d \log n}{n}} + n^{-cd} \right) + C' n e^{-c'n},$$

by Fubini's theorem and Lemma E.37, the triangle inequality (for $|\cdot|$ and for the integral), (E.28), and using $\nu^2 \le \pi^2$ and worst-casing the remaining constants.

To establish the bounds (E.28), we will employ lemma Lemma E.48, which shows that it is sufficient to obtain pointwise control and show a suitable $s$-Lipschitz property for each $i \in [6]$; following the lemma, these properties also imply Lebesgue measurability of the suprema immediately.

**Reduction to product space events.** Fix $\nu$. By the triangle inequality, we have for each $i = 1, \ldots, 6$

$$\left| \mathbb{E}_{\boldsymbol{g}_2}[\Xi_i(\nu, \boldsymbol{g}_1, \boldsymbol{g}_2)] - \mathbb{E}_{\boldsymbol{g}_1, \boldsymbol{g}_2}[\Xi_i(\nu, \boldsymbol{g}_1, \boldsymbol{g}_2)] \right| \le \mathbb{E}_{\boldsymbol{g}_2}\left[ \left| \Xi_i(\nu, \boldsymbol{g}_1, \boldsymbol{g}_2) - \mathbb{E}_{\boldsymbol{g}_1, \boldsymbol{g}_2}[\Xi_i(\nu, \boldsymbol{g}_1, \boldsymbol{g}_2)] \right| \right]. \quad \text{(E.29)}$$

Suppose we can construct $(\boldsymbol{g}_1, \boldsymbol{g}_2)$ events $\mathcal{E}'_i$ such that

1. If $(\boldsymbol{g}_1, \boldsymbol{g}_2) \in \mathcal{E}'_i$, then

$$\left| \Xi_i(\nu, \boldsymbol{g}_1, \boldsymbol{g}_2) - \mathbb{E}_{\boldsymbol{g}_1, \boldsymbol{g}_2}[\Xi_i(\nu, \boldsymbol{g}_1, \boldsymbol{g}_2)] \right| \le C_i \left( \sqrt{\frac{d \log n}{n}} + n^{-c_i d} \right) + C'_i n e^{-c'_i n};$$

2. One has $\mathbb{P}[\mathcal{E}'_i] \ge 1 - C''_i n^{-c''_i d} - C'''_i n e^{-c'''_i n}$.

Then for each such $i$, we can write

$$\mathbb{E}_{\boldsymbol{g}_2}\left[ \left| \Xi_i(\nu, \boldsymbol{g}_1, \boldsymbol{g}_2) - \mathbb{E}_{\boldsymbol{g}_1, \boldsymbol{g}_2}[\Xi_i(\nu, \boldsymbol{g}_1, \boldsymbol{g}_2)] \right| \right]$$

$$= \mathbb{E}_{\boldsymbol{g}_2}\left[ \left( \mathbb{1}_{\mathcal{E}'_i} + \mathbb{1}_{(\mathcal{E}'_i)^{\mathrm{c}}} \right) \left| \Xi_i(\nu, \boldsymbol{g}_1, \boldsymbol{g}_2) - \mathbb{E}_{\boldsymbol{g}_1, \boldsymbol{g}_2}[\Xi_i(\nu, \boldsymbol{g}_1, \boldsymbol{g}_2)] \right| \right]$$

$$\le C_i \left( \sqrt{\frac{d \log n}{n}} + n^{-c_i d} \right) + C'_i n e^{-c'_i n}$$

$$+ \mathbb{E}_{\boldsymbol{g}_2}\left[ \mathbb{1}_{(\mathcal{E}'_i)^{\mathrm{c}}} \left| \Xi_i(\nu, \boldsymbol{g}_1, \boldsymbol{g}_2) - \mathbb{E}_{\boldsymbol{g}_1, \boldsymbol{g}_2}[\Xi_i(\nu, \boldsymbol{g}_1, \boldsymbol{g}_2)] \right| \right] \quad \text{(E.30)}$$

using nonnegativity of the integrand and boundedness of the indicator for $\mathcal{E}'_i$ in the second line. The random variable remaining in the second line is nonnegative, and by Fubini's theorem (with Lemma E.37 for joint integrability) and the Schwarz inequality we have

$$\mathbb{E}_{\boldsymbol{g}_1}\left[ \mathbb{E}_{\boldsymbol{g}_2}\left[ \mathbb{1}_{(\mathcal{E}'_i)^{\mathrm{c}}} \left| \Xi_i(\nu, \boldsymbol{g}_1, \boldsymbol{g}_2) - \mathbb{E}_{\boldsymbol{g}_1, \boldsymbol{g}_2}[\Xi_i(\nu, \boldsymbol{g}_1, \boldsymbol{g}_2)] \right| \right] \right]$$

$$\le \mathbb{E}_{\boldsymbol{g}_1, \boldsymbol{g}_2}\left[ \mathbb{1}_{(\mathcal{E}'_i)^{\mathrm{c}}} \right]^{1/2} \mathbb{E}_{\boldsymbol{g}_1, \boldsymbol{g}_2}\left[ \left( \begin{array}{c} \Xi_i(\nu, \boldsymbol{g}_1, \boldsymbol{g}_2) \\ - \mathbb{E}_{\boldsymbol{g}_1, \boldsymbol{g}_2}[\Xi_i(\nu, \boldsymbol{g}_1, \boldsymbol{g}_2)] \end{array} \right)^2 \right]^{1/2}$$

$$\le C \left( C''_i n^{-c''_i d} + C'''_i n e^{-c'''_i n} \right)^{1/2},$$

where the second line applies Lemma E.37 and the Lyapunov inequality. We can replace this last inequality with one equivalent to the measure bound on $(\mathcal{E}'_i)^{\mathrm{c}}$ using subadditivity of the square root and reducing the constants $c'_i$ and $c''_i$ by a factor of 2. Using this last inequality, Markov's inequality implies for any $t \ge 0$

$$\mathbb{P}\left[ \mathbb{E}_{\boldsymbol{g}_2}\left[ \mathbb{1}_{(\mathcal{E}'_i)^{\mathrm{c}}} \left| \Xi_i(\nu, \boldsymbol{g}_1, \boldsymbol{g}_2) - \mathbb{E}_{\boldsymbol{g}_1, \boldsymbol{g}_2}[\Xi_i(\nu, \boldsymbol{g}_1, \boldsymbol{g}_2)] \right| \right] \ge C n^{-\frac{1}{2} c''_i d} + C' n^{1/2} e^{-\frac{1}{2} c'''_i n} \right]$$

$$\le C n^{-\frac{1}{2} c''_i d} + C' n^{1/2} e^{-\frac{1}{2} c'''_i n},$$

which, together with (E.29) and after worst-casing some exponents and constants, implies that there is a $\boldsymbol{g}_1$ event $\mathcal{E}_i$ that satisfies (the constants $C$ and $C'$ are scoped across properties 1 and 2)

1. If $\boldsymbol{g}_1 \in \mathcal{E}_i$, then

$$\left| \underset{\boldsymbol{g}_2}{\mathbb{E}}[\Xi_i(\nu, \boldsymbol{g}_1, \boldsymbol{g}_2)] - \underset{\boldsymbol{g}_1, \boldsymbol{g}_2}{\mathbb{E}}[\Xi_i(\nu, \boldsymbol{g}_1, \boldsymbol{g}_2)] \right| \le C_i \sqrt{\frac{d \log n}{n}} + (C_i + C) n^{-\frac{1}{2} c_i d}$$
$$+ (C_i' + C') n e^{-\frac{1}{2} \min\{c_i', c_i'''\} n};$$

2. One has $\mathbb{P}[\mathcal{E}_i] \ge 1 - C n^{-\frac{1}{2} c_i'' d} - C' n e^{-\frac{1}{2} c_i''' n}$.

Thus, we can pass from $(\boldsymbol{g}_1, \boldsymbol{g}_2)$ events to $\boldsymbol{g}_1$ events with only a worsening of constants, and it suffices to construct the events $\mathcal{E}_i'$.

Additionally, we can leverage this same framework to pass $\nu$-uniform control from the product space to $\boldsymbol{g}_1$-space. Suppose we can construct $(\boldsymbol{g}_1, \boldsymbol{g}_2)$ events $\mathcal{E}_i'$ such that

1. If $(\boldsymbol{g}_1, \boldsymbol{g}_2) \in \mathcal{E}_i'$, then

$$\forall \nu \in [0, \pi], \left| \Xi_i(\nu, \boldsymbol{g}_1, \boldsymbol{g}_2) - \underset{\boldsymbol{g}_1, \boldsymbol{g}_2}{\mathbb{E}}[\Xi_i(\nu, \boldsymbol{g}_1, \boldsymbol{g}_2)] \right| \le C_i \left( \sqrt{\frac{d \log n}{n}} + n^{-c_i d} \right)$$
$$+ C_i' n e^{-c_i' n};$$

2. One has $\mathbb{P}[\mathcal{E}_i'] \ge 1 - C_i'' n^{-c_i'' d} - C_i''' n e^{-c_i''' n}$.

Then following (E.30), we can assert

$$\forall \nu \in [0, \pi], \underset{\boldsymbol{g}_2}{\mathbb{E}}\left[ \left| \Xi_i(\nu, \boldsymbol{g}_1, \boldsymbol{g}_2) - \underset{\boldsymbol{g}_1, \boldsymbol{g}_2}{\mathbb{E}}[\Xi_i(\nu, \boldsymbol{g}_1, \boldsymbol{g}_2)] \right| \right]$$
$$\le C_i \left( \sqrt{\frac{d \log n}{n}} + n^{-c_i d} \right) + C_i' n e^{-c_i' n} + \underset{\boldsymbol{g}_2}{\mathbb{E}}\left[ \mathbb{1}_{(\mathcal{E}_i')^{\mathsf{c}}} \left| \Xi_i(\nu, \boldsymbol{g}_1, \boldsymbol{g}_2) - \underset{\boldsymbol{g}_1, \boldsymbol{g}_2}{\mathbb{E}}[\Xi_i(\nu, \boldsymbol{g}_1, \boldsymbol{g}_2)] \right| \right].$$

To get uniform control of this last random variable, we can use Lemma E.37, which tells us that we have a bound

$$\forall \nu \in [0, \pi], \underset{\boldsymbol{g}_2}{\mathbb{E}}\left[ \left| \Xi_i(\nu, \boldsymbol{g}_1, \boldsymbol{g}_2) - \underset{\boldsymbol{g}_1, \boldsymbol{g}_2}{\mathbb{E}}[\Xi_i(\nu, \boldsymbol{g}_1, \boldsymbol{g}_2)] \right| \right]$$
$$\le C_i \left( \sqrt{\frac{d \log n}{n}} + n^{-c_i d} \right) + C_i' n e^{-c_i' n} + \underset{\boldsymbol{g}_2}{\mathbb{E}}\left[ \mathbb{1}_{(\mathcal{E}_i')^{\mathsf{c}}} f_i(\boldsymbol{g}_1, \boldsymbol{g}_2) \right], \tag{E.31}$$

where $f_i$ is in $L^4(\mathbb{R}^n \times \mathbb{R}^n)$, and has $L^4$ norm bounded by an absolute constant $C_i > 0$. Then Fubini's theorem and the Schwarz inequality allow us to assert

$$\underset{\boldsymbol{g}_1, \boldsymbol{g}_2}{\mathbb{E}}\left[ \mathbb{1}_{(\mathcal{E}_i')^{\mathsf{c}}} f_i(\boldsymbol{g}_1, \boldsymbol{g}_2) \right] \le C_i \underset{\boldsymbol{g}_1, \boldsymbol{g}_2}{\mathbb{E}}\left[ \mathbb{1}_{(\mathcal{E}_i')^{\mathsf{c}}} \right]^{1/2},$$

which can be controlled exactly as in the pointwise control argument. In particular, an application of Markov's inequality gives

$$\mathbb{P}\left[ \underset{\boldsymbol{g}_2}{\mathbb{E}}\left[ \mathbb{1}_{(\mathcal{E}_i')^{\mathsf{c}}} f_i(\boldsymbol{g}_1, \boldsymbol{g}_2) \right] \ge C n^{-\frac{1}{2} c_i'' d} + C' n^{1/2} e^{-\frac{1}{2} c_i''' n} \right] \le C n^{-\frac{1}{2} c_i'' d} + C' n^{1/2} e^{-\frac{1}{2} c_i''' n},$$

so that, returning to (E.31), we have uniform control of the quantity $|\mathbb{E}_{\boldsymbol{g}_2}[\Xi_i(\nu, \boldsymbol{g}_1, \boldsymbol{g}_2)] - \mathbb{E}_{\boldsymbol{g}_1, \boldsymbol{g}_2}[\Xi_i(\nu, \boldsymbol{g}_1, \boldsymbol{g}_2)]|$ on an event of appropriately high probability. In particular, we have incurred only losses in the constants compared to the pointwise case.

**Approach to Lipschitz estimates.** We will use this framework for controlling the $\Xi_1$ and $\Xi_5$ terms only. Accordingly, the sections for those terms below will produce results of the following type, for absolute constants $c_i, c_i', c_i'', c_i''', C_i, C_i', C_i'', C_i''', C_i'''' > 0$ for $i = 1, 2$, and parameters $d \geq 1, \delta > 0$ such that $d$ and $\delta$ are larger than (separate) absolute constants and $n$ satisfies certain conditions involving $d$:

1. For each $\nu \in [0, \pi]$ fixed, with probability at least $1 - C_1''' n^{-c_1'' d} - C_1'''' n e^{-c_1''' n}$, we have that

$$\left| \mathbb{E}_{\boldsymbol{g}_2} [\Xi_i(\nu, \boldsymbol{g}_1, \boldsymbol{g}_2) - \mathbb{E}[\Xi_i(\nu, \boldsymbol{g}_1, \boldsymbol{g}_2)]] \right| \leq C_1 \sqrt{d \log n / n} + C_1' n^{-c_1 d} + C_1'' n e^{-c_1' n};$$

2. With probability at least $1 - C_2'' e^{-c_2' n} - C_2''' n^{-\delta}$, we have that $|\mathbb{E}_{\boldsymbol{g}_2}[\Xi_i(\nu, \boldsymbol{g}_1, \boldsymbol{g}_2) - \mathbb{E}[\Xi_i(\nu, \boldsymbol{g}_1, \boldsymbol{g}_2)]]|$ is $(C_2 + C_2' n^{1+\delta})$-Lipschitz.

We show here that we can use these properties to obtain uniform concentration of the relevant quantities. Write $M = C_1 \sqrt{d \log n / n} + C_1' n^{-c_1 d} + C_1'' n e^{-c_1' n}$; we are interested in showing that uniform bounds of sizes close to $M$ hold with probability not much smaller than that of the pointwise bounds. By Lemma E.48, it follows from the assumed properties that for any $0 < \varepsilon < 1$ one has

$$\mathbb{P}\left[ \sup_{\nu \in [0,\pi]} \left| \mathbb{E}_{\boldsymbol{g}_2}[\Xi_i(\nu, \boldsymbol{g}_1, \boldsymbol{g}_2) - \mathbb{E}[\Xi_i(\nu, \boldsymbol{g}_1, \boldsymbol{g}_2)]] \right| \leq M + \varepsilon \left( C_2 + C_2' n^{1+\delta} \right) \right]$$

$$\geq 1 - \left( C_1''' n^{-c_1'' d} + C_1'''' n e^{-c_1''' n} \right) K \varepsilon^{-1} - \left( C_2'' e^{-c_2' n} + C_2''' n^{-\delta} \right),$$

where $K > 0$ is an absolute constant. To make the RHS of the bound on the supremum of size comparable to $M$, it suffices to choose $\varepsilon = C_1 \sqrt{d \log n / n} / (C_2 + C_2' n^{1+\delta})$. We have $C_2 + C_2' n^{1+\delta} \leq K' n^{1+\delta}$ for $K' > 0$ an absolute constant, and so we have $\varepsilon^{-1} \leq K' n^{3/2+\delta}$ for $K' > 0$ another absolute constant. This gives

$$\left( C_1''' n^{-c_1'' d} + C_1'''' n e^{-c_1''' n} \right) \varepsilon^{-1} \leq K' n^{3/2+\delta} \left( C_1''' e^{-c_1'' d \log n} + C_1'''' e^{-c_1''' / 2n} \right)$$

$$\leq K' n^{3/2+\delta} e^{-c_1'' d \log n}$$

$$\leq K' n^{-c_1'' d / 2},$$

where $K' > 0$ is an absolute constant whose value changes from line to line; and where the first inequality assumes that $n \geq (2/c_1''') \log n$, the second inequality assumes that $n \geq (2 c_1'' / c_1''') d \log n$, and the third assumes that $\delta \leq c_1'' d / 2 - 3/2$. Choosing $d$ so that the value $c_1'' d / 2 - 3/2$ is larger than the minimum value for $\delta$ (i.e., larger than an absolute constant), then choosing $\delta = c_1'' d / 2 - 3/2$, and finally choosing $d \geq 6/c_1''$, we obtain

$$\mathbb{P}\left[ \sup_{\nu \in [0,\pi]} \left| \mathbb{E}_{\boldsymbol{g}_2}[\Xi_i(\nu, \boldsymbol{g}_1, \boldsymbol{g}_2) - \mathbb{E}[\Xi_i(\nu, \boldsymbol{g}_1, \boldsymbol{g}_2)]] \right| \leq 2M \right] \geq 1 - K n^{-c_1'' d / 2} - C_2'' e^{-c_2' n} - C_2''' n^{-c_1'' d / 4},$$

where $K > 0$ is an absolute constant, which is an acceptable level of uniformization.

**Completing the proof.** To obtain the desired control, we apply the uniform framework for the terms $\Xi_i$, $i = 2, 3, 4, 6$; and the pointwise with Lipschitz control framework for the terms $\Xi_i$, $i = 1, 5$. We also establish high probability control of the zero-order term in Lemma E.38. The events we need for the pointwise framework terms are constructed in Lemmas E.39, E.40, E.44 and E.45. The events we need for the uniform framework are constructed in Lemmas E.41 to E.43 and E.46. Because $n$ and $d$ are chosen appropriately by our hypotheses here, we can invoke each of these lemmas to construct the necessary sub-events and obtain an event $\mathcal{E}$ which satisfies

1. One has

$$\forall \nu \in [0, \pi], \quad \left| \mathbb{E}_{\boldsymbol{g}_2}[Y_\nu] - \mathbb{E}_{\boldsymbol{g}_1, \boldsymbol{g}_2}[Y_\nu] \right| \leq C\nu^2 \left( \sqrt{\frac{d \log n}{n}} + n^{-cd} \right) + C' n e^{-c' n}$$

if $\boldsymbol{g}_1 \in \mathcal{E}$;

2. One has

$$\mathbb{P}[\mathcal{E}] \geq 1 - C'' n^{-c'' d} - C''' n e^{-c''' n}.$$

We can adjust $d$ and $n$ slightly to obtain an event with the properties claimed in the statement of the lemma. Indeed, choosing $n$ to be larger than an absolute constant multiple of $\log n$, we can obtain $C' n e^{-c' n} \leq C' e^{-c' n/2}$ and $C''' n e^{-c''' n} \leq C''' e^{-c''' n/2}$; choosing $n$ to be larger than an absolute constant multiple of $d \log n$, we can obtain $C'' n^{-c'' d} + C''' e^{-c''' n/2} \leq 2 C'' n^{-c'' d}$; and choosing $d$ to be larger than an absolute constant, we can assert $\sqrt{d \log n / n} + n^{-cd} \leq 2\sqrt{d \log n / n}$. This turns the guarantees of $\mathcal{E}$ into the guarantees claimed in the statement of the lemma, and completes the proof.

$\square$

### E.3.3 PROVING LEMMA E.7

**Lemma E.14.** *One has bounds*

$$1 - \frac{\nu^2}{2} \leq \cos \varphi(\nu) \leq 1 - c\nu^2, \qquad \nu \in [0, \pi].$$

*Proof.* Write $f(\nu) = \cos \varphi(\nu) = \cos \nu + \pi^{-1}(\sin \nu - \nu \cos \nu)$, where the last equality follows from Lemma E.2. We start by obtaining quadratic bounds on $f(\nu)$ for $\nu \in [0, 0.1]$. In particular, we will show

$$1 - \tfrac{1}{2}\nu^2 \leq f(\nu) \leq 1 - \tfrac{1}{4}\nu^2, \quad \nu \in [0, 0.1]. \tag{E.32}$$

We readily calculate

$$f'(\nu) = -\sin \nu + \pi^{-1}\nu \sin \nu,$$
$$f''(\nu) = -\cos \nu + \pi^{-1}(\nu \cos \nu + \sin \nu).$$

Taylor expanding at $\nu = 0$ gives

$$1 + \frac{\inf_{t \in [0,0.1]} f''(t)}{2}\nu^2 \leq f(\nu) \leq 1 + \frac{\sup_{t \in [0,0.1]} f''(t)}{2}\nu^2.$$

We have $f''(0) = -1$, and $\sin \nu \leq \sin 0.1$ on our interval of interest by monotonicity. The derivative of $\nu \cos \nu$ is $\cos \nu - \nu \sin \nu$; $\nu \sin \nu$ is increasing as the product of two increasing functions (given $\nu \leq 0.1$), and one checks that $\cos(0.1) - 0.1 \sin(0.1) > 0$; therefore $\nu \cos \nu \leq 0.1 \cos(0.1)$ on our domain of interest. One checks numerically

$$-\cos(0.1) + \pi^{-1}(0.1 \cos(0.1) + \sin(0.1)) < -\tfrac{1}{2} < 0,$$

and this establishes $f(\nu) \leq 1 - \tfrac{1}{4}\nu^2$ on $[0, 0.1]$. If $\nu \leq \pi/2$, we have $\cos \geq 0$ and $\sin \geq 0$, so that $\nu \cos \nu + \sin \nu \geq 0$ on this domain. This implies $f''(\nu) \geq -\cos \nu \geq -1$ for $0 \leq \nu \leq \pi/2$, which proves $\inf_{t \in [0,\pi/2]} f''(t) = -1$, and establishes the lower bound on $[0, \pi/2]$.

To obtain (possibly) looser bounds on $[0, \pi]$, we use a bootstrapping approach. The lower bound is more straightforward; to assert the lower bound on $[0, \pi]$, we evaluate constants numerically to find that the lower bound's value at $\pi/2$ is $1 - \pi^2/8 < 0$, and given that $f \geq 0$ by Lemma E.5 and the concave quadratic bound is maximized at $\nu = 0$, it follows that the bound holds on the entire interval.

For bootstrapping the upper bound, we note that the equation

$$f'(\nu) = -\sin \nu + \pi^{-1}\nu \sin \nu = \sin \nu \left(\frac{\nu}{\pi} - 1\right)$$

shows immediately that $f$ is a strictly decreasing function of $\nu$ on $(0, \pi)$. Therefore $f(\nu) \leq f(0.1)$ on $[0.1, \pi]$, and so the quadratic function $\nu \mapsto 1 - \pi^{-2}(1 - f(0.1))\nu^2$, which is lower bounded by $1 - \nu^2/4$ on $[0, \pi]$ by the fact that both concave quadratic functions are maximized at $0$ and the verification $1 - \pi^2/4 < 0 \leq f(0.1)$, is an upper bound for $f$ on all of $[0, \pi]$; so the claim holds with $c = \pi^{-2}(1 - f(0.1))$. $\square$

**Lemma E.15.** *There exist absolute constants $c, C, C', C'' > 0$ such that if $n \geq C \log n$, then one has*

$$\left| \mathop{\mathbb{E}}_{\boldsymbol{g}_1, \boldsymbol{g}_2} [X_\nu] - \cos \varphi(\nu) \right| \leq C' e^{-cn} + C'' \nu^2 / n.$$

*Proof.* Write $h(\nu) = \cos \varphi(\nu) - \mathbb{E}[X_\nu]$. By Lemmas E.24 and E.25, we have a second-order Taylor formula

$$h(\nu) = h(0) + \int_0^\nu \left( h'(0) + \int_0^t h''(s) \, \mathrm{d}s \right) \mathrm{d}t.$$

We calculate $h'(0) = 0$, since $\mathbb{E}[\langle \boldsymbol{v}_0, \dot{\boldsymbol{v}}_0 \rangle] = \mathbb{E}[\langle \sigma(\boldsymbol{g}_1), \boldsymbol{g}_2 \rangle] = 0$, and $\boldsymbol{P}_{\boldsymbol{v}_0}^\perp \boldsymbol{v}_0 = \boldsymbol{0}$. We also have $h(0) = \mathbb{E}[\|\boldsymbol{v}_0\|_2^2] - \mathbb{E}[\mathbb{1}_{\mathcal{E}_m}] = \mu(\mathcal{E}_m^{\mathsf{c}})$ (writing $m = 1$), so this formula yields

$$|h(\nu)| \leq \mu(\mathcal{E}_m^{\mathsf{c}}) + \frac{\nu^2}{2} \operatorname*{ess\,sup}_{\nu' \in [0, \pi]} |h''(\nu')|,$$

and we see that it suffices to bound $h''$. We will use the (Lebesgue-a.e.) expression

$$|h''(\nu)| = \left| \mathbb{E}[\langle \dot{\boldsymbol{v}}_\nu, \dot{\boldsymbol{v}}_0 \rangle] - \mathbb{E}\left[ \mathbb{1}_{\mathcal{E}_m} \left\langle \frac{1}{\|\boldsymbol{v}_\nu\|_2} \left( \boldsymbol{I} - \frac{\boldsymbol{v}_\nu \boldsymbol{v}_\nu^*}{\|\boldsymbol{v}_\nu\|_2^2} \right) \dot{\boldsymbol{v}}_\nu, \frac{1}{\|\boldsymbol{v}_0\|_2} \left( \boldsymbol{I} - \frac{\boldsymbol{v}_0 \boldsymbol{v}_0^*}{\|\boldsymbol{v}_0\|_2^2} \right) \dot{\boldsymbol{v}}_0 \right\rangle \right] \right|.$$

Distributing over the inner product and applying rotational invariance to combine the two cross terms, then using the triangle inequality, we obtain the bound

$$|h''(\nu)| \leq \underbrace{\left| \mathbb{E}[\langle \dot{\boldsymbol{v}}_\nu, \dot{\boldsymbol{v}}_0 \rangle] - \mathbb{E}\left[ \mathbb{1}_{\mathcal{E}_m} \frac{\langle \dot{\boldsymbol{v}}_0, \dot{\boldsymbol{v}}_\nu \rangle}{\|\boldsymbol{v}_0\|_2 \|\boldsymbol{v}_\nu\|_2} \right] \right|}_{\Xi_1(\nu)} + \underbrace{\left| 2 \mathbb{E}\left[ \mathbb{1}_{\mathcal{E}_m} \frac{\langle \dot{\boldsymbol{v}}_0, \boldsymbol{v}_\nu \rangle \langle \boldsymbol{v}_\nu, \dot{\boldsymbol{v}}_\nu \rangle}{\|\boldsymbol{v}_0\|_2 \|\boldsymbol{v}_\nu\|_2^3} \right] \right|}_{\Xi_2(\nu)}$$

$$+ \underbrace{\left| \mathbb{E}\left[ \mathbb{1}_{\mathcal{E}_m} \frac{\langle \dot{\boldsymbol{v}}_0, \boldsymbol{v}_0 \rangle \langle \boldsymbol{v}_0, \boldsymbol{v}_\nu \rangle \langle \boldsymbol{v}_\nu, \dot{\boldsymbol{v}}_\nu \rangle}{\|\boldsymbol{v}_0\|_2^3 \|\boldsymbol{v}_\nu\|_2^3} \right] \right|}_{\Xi_3(\nu)}.$$

We proceed by giving magnitude bounds for $\Xi_i(\nu)$, $i = 1, 2, 3$. Because we are working with expectations, it suffices to fix one value $\nu \in [0, \pi]$ and prove pointwise $\nu$-independent bounds; we will exploit this in the sequel to easily define extra good events without having to uniformize, and we will generally suppress the notational dependence of $\Xi_i$ on $\nu$ as a result. We will also repeatedly use the fact that we have $\mu(\mathcal{E}_1^{\mathsf{c}}) \leq C n e^{-cn}$ for some absolute constants $c, C > 0$ by Lemma E.16. We will accrue a large number of additive $C/n$ and $C' n^{pm} e^{-cn}$ errors as we bound the $\Xi_i$ terms; at the end of the proof we will worst-case the constants in each additive error and assert a bound of the form claimed.

$\Xi_1$ **control.** Let $\mathcal{E} = \{\|\dot{\boldsymbol{v}}_\nu\|_2 \leq 2\} \cap \{\|\dot{\boldsymbol{v}}_0\|_2 \leq 2\}$. By Lemma E.17 and a union bound, we have $\mu(\mathcal{E}^{\mathsf{c}}) \leq C e^{-cn}$. Define an event $\mathcal{E}_1 = \mathcal{E}_m \cap \mathcal{E}$. The first step is to pass to the control of

$$\widetilde{\Xi}_1 := \mathbb{E}\left[ \mathbb{1}_{\mathcal{E}_1} \langle \dot{\boldsymbol{v}}_\nu, \dot{\boldsymbol{v}}_0 \rangle \left( 1 - \frac{1}{\|\boldsymbol{v}_0\|_2 \|\boldsymbol{v}_\nu\|_2} \right) \right].$$

The triangle inequality gives

$$\left| \Xi_1 - \widetilde{\Xi}_1 \right| \leq \left| \mathbb{E}[\mathbb{1}_{\mathcal{E}_1^{\mathsf{c}}} \langle \dot{\boldsymbol{v}}_\nu, \dot{\boldsymbol{v}}_0 \rangle] \right| + \left| \mathbb{E}\left[ \mathbb{1}_{\mathcal{E}_m \backslash \mathcal{E}} \left\langle \frac{\dot{\boldsymbol{v}}_\nu}{\|\boldsymbol{v}_\nu\|_2}, \frac{\dot{\boldsymbol{v}}_0}{\|\boldsymbol{v}_0\|_2} \right\rangle \right] \right|$$

The first term is readily controlled from two applications of the Schwarz inequality, a union bound, and rotational invariance together with Lemma E.29:

$$\left| \mathbb{E}[\mathbb{1}_{\mathcal{E}_1^{\mathsf{c}}} \langle \dot{\boldsymbol{v}}_\nu, \dot{\boldsymbol{v}}_0 \rangle] \right| \leq \mathbb{E}[\mathbb{1}_{\mathcal{E}_1^{\mathsf{c}}}]^{1/2} \mathbb{E}[\|\dot{\boldsymbol{v}}_\nu\|_2^4]^{1/4} \mathbb{E}[\|\dot{\boldsymbol{v}}_0\|_2^4]^{1/4}$$

$$\leq \left( \mu(\mathcal{E}_m^{\mathsf{c}}) + C e^{-cn} \right)^{1/2} \mathbb{E}[\|\dot{\boldsymbol{v}}_0\|_2^4]^{1/2}$$

$$\leq \left( C n e^{-cn} + C' e^{-c'n} \right)^{1/2} \left( 1 + \frac{C''}{n} \right)^{1/2}$$

$$\leq C n^{1/2} e^{-cn},$$

where in the last line we require $n$ to be at least the value of a large absolute constant. The calculation is similar for the normalized term, except we also apply the definition of $\mathcal{E}_m$ to get some extra cancellation:

$$
\left| \mathbb{E}\left[ \mathbb{1}_{\mathcal{E}_m \setminus \mathcal{E}} \frac{\langle \dot{\boldsymbol{v}}_\nu, \dot{\boldsymbol{v}}_0 \rangle}{\|\boldsymbol{v}_\nu\|_2 \|\boldsymbol{v}_0\|_2} \right] \right| \leq \mathbb{E}\left[ \mathbb{1}_{\mathcal{E}_m \setminus \mathcal{E}} \frac{|\langle \dot{\boldsymbol{v}}_\nu, \dot{\boldsymbol{v}}_0 \rangle|}{\|\boldsymbol{v}_\nu\|_2 \|\boldsymbol{v}_0\|_2} \right] \leq 4\mathbb{E}\left[ \mathbb{1}_{\mathcal{E}_m \setminus \mathcal{E}} |\langle \dot{\boldsymbol{v}}_\nu, \dot{\boldsymbol{v}}_0 \rangle| \right]
$$
$$
\leq 4\mathbb{E}[\mathbb{1}_{\mathcal{E}^c} |\langle \dot{\boldsymbol{v}}_\nu, \dot{\boldsymbol{v}}_0 \rangle|]
$$
$$
\leq 4\mathbb{E}[\mathbb{1}_{\mathcal{E}^c}]^{1/2} \mathbb{E}\left[ \|\dot{\boldsymbol{v}}_\nu\|_2^4 \right]^{1/4} \mathbb{E}\left[ \|\dot{\boldsymbol{v}}_0\|_2^4 \right]^{1/4}
$$
$$
\leq Ce^{-cn}\left( 1 + \frac{C'}{n} \right)^{1/2} \leq Ce^{-cn},
$$

where in the last line we apply our bounds from the first term and use $n \geq 1$ to obtain the final inequality. Next, Taylor expansion of the smooth convex function $x \mapsto x^{-1/2}$ on the domain $x > 0$ about the point $x = 1$ gives

$$
x^{-1/2} = 1 - \frac{1}{2}(x-1) + \frac{3}{4} \int_1^x (x-t) t^{-5/2} \, \mathrm{d}t. \tag{E.33}
$$

Given that $\mathcal{E}_m$ guarantees $\|\boldsymbol{v}_\nu\|_2 \geq \frac{1}{2}$, we can apply this to get a bound

$$
\mathbb{1}_{\mathcal{E}_1}\left( 1 - \frac{1}{\|\boldsymbol{v}_0\|_2 \|\boldsymbol{v}_\nu\|_2} \right)
$$
$$
= \mathbb{1}_{\mathcal{E}_1}\left( \frac{1}{2}\left( \|\boldsymbol{v}_0\|_2^2 \|\boldsymbol{v}_\nu\|_2^2 - 1 \right) - \frac{3}{4} \int_1^{\|\boldsymbol{v}_0\|_2^2 \|\boldsymbol{v}_\nu\|_2^2} \left( \|\boldsymbol{v}_0\|_2^2 \|\boldsymbol{v}_\nu\|_2^2 - t \right) t^{-5/2} \, \mathrm{d}t \right).
$$

On $\mathcal{E}'$, we also have $\|\boldsymbol{v}_0\|_2^{-2} \|\boldsymbol{v}_\nu\|_2^{-2} \leq 2^4$, so we can control the integral residual as

$$
0 \leq \mathbb{1}_{\mathcal{E}_1} \frac{3}{4} \int_1^{\|\boldsymbol{v}_0\|_2^2 \|\boldsymbol{v}_\nu\|_2^2} \left( \|\boldsymbol{v}_0\|_2^2 \|\boldsymbol{v}_\nu\|_2^2 - t \right) t^{-5/2} \, \mathrm{d}t \leq \mathbb{1}_{\mathcal{E}_1} 384 \left( \|\boldsymbol{v}_0\|_2^2 \|\boldsymbol{v}_\nu\|_2^2 - 1 \right)^2,
$$

where we replace the tighter bound that we get in the case $\|\boldsymbol{v}_0\|_2^2 \|\boldsymbol{v}_\nu\|_2^2 \geq 1$ with the worst-case bound from the other case. This gives bounds

$$
\mathbb{1}_{\mathcal{E}_1}\left( \frac{1}{2}\left( \|\boldsymbol{v}_0\|_2^2 \|\boldsymbol{v}_\nu\|_2^2 - 1 \right) - 384 \left( \|\boldsymbol{v}_0\|_2^2 \|\boldsymbol{v}_\nu\|_2^2 - 1 \right)^2 \right) \leq \mathbb{1}_{\mathcal{E}_1}\left( 1 - \frac{1}{\|\boldsymbol{v}_0\|_2 \|\boldsymbol{v}_\nu\|_2} \right)
$$
$$
\leq \mathbb{1}_{\mathcal{E}_1} \frac{1}{2}\left( \|\boldsymbol{v}_0\|_2^2 \|\boldsymbol{v}_\nu\|_2^2 - 1 \right).
$$

Given that $\|\dot{\boldsymbol{v}}_\nu\|_2 \leq 2$ on $\mathcal{E}''$, it follows $|\langle \dot{\boldsymbol{v}}_0, \dot{\boldsymbol{v}}_\nu \rangle| \leq 4$ on $\mathcal{E}_1$, so that $\langle \dot{\boldsymbol{v}}_0, \dot{\boldsymbol{v}}_\nu \rangle + 4 \geq 0$ here. Writing

$$
\mathbb{1}_{\mathcal{E}_1} \langle \dot{\boldsymbol{v}}_0, \dot{\boldsymbol{v}}_\nu \rangle \left( 1 - \frac{1}{\|\boldsymbol{v}_0\|_2 \|\boldsymbol{v}_\nu\|_2} \right)
$$
$$
= \mathbb{1}_{\mathcal{E}_1}\left[ \left( \langle \dot{\boldsymbol{v}}_0, \dot{\boldsymbol{v}}_\nu \rangle + 4 \right)\left( 1 - \frac{1}{\|\boldsymbol{v}_0\|_2 \|\boldsymbol{v}_\nu\|_2} \right) - 4\left( 1 - \frac{1}{\|\boldsymbol{v}_0\|_2 \|\boldsymbol{v}_\nu\|_2} \right) \right],
$$

we can apply nonnegativity to obtain upper and lower bounds

$$
\widetilde{\widetilde{\Xi}}_1 \leq \mathbb{E}\left[ \mathbb{1}_{\mathcal{E}_1} \langle \dot{\boldsymbol{v}}_0, \dot{\boldsymbol{v}}_\nu \rangle \left( \frac{1}{2}\left( \|\boldsymbol{v}_0\|_2^2 \|\boldsymbol{v}_\nu\|_2^2 - 1 \right) + 4C \left( \|\boldsymbol{v}_0\|_2^2 \|\boldsymbol{v}_\nu\|_2^2 - 1 \right)^2 \right) \right];
$$
$$
\widetilde{\widetilde{\Xi}}_1 \geq \mathbb{E}\left[ \mathbb{1}_{\mathcal{E}_1} \langle \dot{\boldsymbol{v}}_0, \dot{\boldsymbol{v}}_\nu \rangle \left( \frac{1}{2}\left( \|\boldsymbol{v}_0\|_2^2 \|\boldsymbol{v}_\nu\|_2^2 - 1 \right) - 5C \left( \|\boldsymbol{v}_0\|_2^2 \|\boldsymbol{v}_\nu\|_2^2 - 1 \right)^2 \right) \right],
$$

where $C = 384$.

We continue with bounding the quadratic term arising in the previous equation. We have

$$
\begin{aligned}
\left| \mathbb{E}\Big[ \mathbb{1}_{\mathcal{E}_1} \langle \dot{\boldsymbol{v}}_0, \dot{\boldsymbol{v}}_\nu \rangle \big( \|\boldsymbol{v}_0\|_2^2 \|\boldsymbol{v}_\nu\|_2^2 - 1 \big)^2 \Big] \right| &\le 4\mathbb{E}\Big[ \big( \|\boldsymbol{v}_0\|_2^2 \|\boldsymbol{v}_\nu\|_2^2 - 1 \big)^2 \Big] \\
&= 4\mathbb{E}\big[ \|\boldsymbol{v}_0\|_2^4 \|\boldsymbol{v}_\nu\|_2^4 - 2\|\boldsymbol{v}_0\|_2^2 \|\boldsymbol{v}_\nu\|_2^2 + 1 \big] \\
&\le 4\big( 1 - 2\mathbb{E}[\|\boldsymbol{v}_0\|_2^2 \|\boldsymbol{v}_\nu\|_2^2] + \mathbb{E}[\|\boldsymbol{v}_0\|_2^8] \big) \\
&\le 4\left( 1 - 2(1 - (Cn^{-1} + C'e^{-cn}))^2 + \left( 1 + \frac{C''}{n} \right) \right) \\
&\le Cn^{-1}e^{-cn} + C'e^{-c'n} + \frac{C''}{n}.
\end{aligned}
$$

The first inequality applies the triangle inequality for the integral, the definition of $\mathcal{E}_1$ and Cauchy-Schwarz, then drops the indicator for $\mathcal{E}_1$ because the remaining terms are nonnegative; the second line is just distributing; the third line rearranges and applies the Schwarz inequality; and the fourth inequality applies Jensen's inequality and Lemma E.18 to control the second term (to apply this lemma, we need to choose $n$ larger than an absolute constant; we assume this is done), and Lemma E.29 to control the third term. Since $n \ge 1$, this gives a $C/n + C'e^{-cn}$ bound on the quadratic term.

Next is the linear term; our first step will be to get rid of the indicator. By the triangle inequality, it suffices to get control of the corresponding term with the indicator for $\mathcal{E}_1^{\mathsf{c}}$ instead; we control it as follows:

$$
\begin{aligned}
\big| \mathbb{E}\big[ \mathbb{1}_{\mathcal{E}_1^{\mathsf{c}}} \langle \dot{\boldsymbol{v}}_0, \dot{\boldsymbol{v}}_\nu \rangle \big( \|\boldsymbol{v}_0\|_2^2 \|\boldsymbol{v}_\nu\|_2^2 - 1 \big) \big] \big| & \\
&\hspace{-9em} \le \mathbb{E}\big[ \mathbb{1}_{\mathcal{E}_1^{\mathsf{c}}} \big]^{1/2} \mathbb{E}\Big[ \langle \dot{\boldsymbol{v}}_0, \dot{\boldsymbol{v}}_\nu \rangle^2 \big( \|\boldsymbol{v}_0\|_2^2 \|\boldsymbol{v}_\nu\|_2^2 - 1 \big)^2 \Big]^{1/2} \\
&\hspace{-9em} \le \big( Cne^{-cn} + C'e^{-c'n} \big)^{1/2} \mathbb{E}\Big[ \|\dot{\boldsymbol{v}}_0\|_2^2 \|\dot{\boldsymbol{v}}_\nu\|_2^2 \big( \|\boldsymbol{v}_0\|_2^2 \|\boldsymbol{v}_\nu\|_2^2 - 1 \big)^2 \Big]^{1/2} \\
&\hspace{-9em} \le \big( Cne^{-cn} + C'e^{-c'n} \big)^{1/2} \mathbb{E}\big[ \|\dot{\boldsymbol{v}}_0\|_2^2 \|\dot{\boldsymbol{v}}_\nu\|_2^2 \|\boldsymbol{v}_0\|_2^4 \|\boldsymbol{v}_\nu\|_2^4 + \|\dot{\boldsymbol{v}}_0\|_2^2 \|\dot{\boldsymbol{v}}_\nu\|_2^2 \big]^{1/2} \\
&\hspace{-9em} \le \big( Cne^{-cn} + C'e^{-c'n} \big)^{1/2} \Big( \mathbb{E}\big[ \|\dot{\boldsymbol{v}}_0\|_2^8 \big]^{1/4} \mathbb{E}\big[ \|\boldsymbol{v}_0\|_2^{16} \big]^{1/4} + \mathbb{E}\big[ \|\dot{\boldsymbol{v}}_0\|_2^4 \big]^{1/2} \Big) \\
&\hspace{-9em} \le \big( Cne^{-cn} + C'e^{-c'n} \big)^{1/2} \left( \left( 1 + \frac{C_1}{n} \right)^{1/4} \left( 1 + \frac{C_2}{n} \right)^{1/4} + \left( 1 + \frac{C_3}{n} \right)^{1/2} \right) \\
&\hspace{-9em} \le Cn^{1/2}e^{-cn} + C'e^{-c'n}.
\end{aligned}
$$

The first line is the Schwarz inequality; the second line is the good event measure bound and Cauchy-Schwarz; the third line distributes and drops the cross term, given that all factors are nonnegative; the fourth line applies subadditivity of the square root function, then the Schwarz inequality to the resulting separate terms; the fifth line applies Lemma E.29; and the last line again uses square root subadditivity and treats the remaining terms as multiplicative constants, since $n \ge 1$. Therefore passing to the linear term without the indicator incurs only an additional exponential factor. Proceeding, we drop the indicator and distribute to get for the linear term

$$
\mathbb{E}\big[ \langle \dot{\boldsymbol{v}}_0, \dot{\boldsymbol{v}}_\nu \rangle \big( \|\boldsymbol{v}_0\|_2^2 \|\boldsymbol{v}_\nu\|_2^2 - 1 \big) \big] = \mathbb{E}\big[ \langle \dot{\boldsymbol{v}}_0, \dot{\boldsymbol{v}}_\nu \rangle \|\boldsymbol{v}_0\|_2^2 \|\boldsymbol{v}_\nu\|_2^2 \big] - \mathbb{E}[\langle \dot{\boldsymbol{v}}_0, \dot{\boldsymbol{v}}_\nu \rangle];
$$

it is of interest to apply Lemma E.30 to these two terms to get the proper cancellation, and for this we just need to check that the coordinates of each factor in the product have subexponential moment growth with the proper rate. For even powers of $\ell^2$ norms of $\boldsymbol{v}_\nu$, this follows immediately from Lemma G.11 after scaling by $\sqrt{2/n}$; for the inner product term, the coordinate functions are $\dot{\sigma}(g_{1i})g_{2i}\dot{\sigma}(g_{1i}\cos\nu + g_{2i}\sin\nu)(g_{2i}\cos\nu - g_{1i}\sin\nu)$, and we have from the Schwarz inequality and rotational invariance

$$
\mathbb{E}\big[ |\dot{\sigma}(g_{1i})g_{2i}\dot{\sigma}(g_{1i}\cos\nu + g_{2i}\sin\nu)(g_{2i}\cos\nu - g_{1i}\sin\nu)|^k \big] \le \mathbb{E}\big[ \dot{\sigma}(g_{1i})g_{2i}^{2k} \big],
$$

which has subexponential moment growth with rate $Cn^{-1}$ by Lemma E.17 and Lemma G.11 after rescaling by $\sqrt{2/n}$. These formulas also show that when $k = 1$, we have a bound of precisely $n^{-1}$. This makes Lemma E.30 applicable, so we can assert bounds

$$
\left| \mathbb{E}\big[ \langle \dot{\boldsymbol{v}}_0, \dot{\boldsymbol{v}}_\nu \rangle \big( \|\boldsymbol{v}_0\|_2^2 \|\boldsymbol{v}_\nu\|_2^2 - 1 \big) \big] - \left( n^3 \mathbb{E}[(\dot{\boldsymbol{v}}_0)_1 (\dot{\boldsymbol{v}}_\nu)_1] \mathbb{E}\big[ \sigma(g_{11})^2 \big]^2 - n\mathbb{E}[(\dot{\boldsymbol{v}}_0)_1 (\dot{\boldsymbol{v}}_\nu)_1] \right) \right| \le \frac{C}{n}
$$

Because $\mathbb{E}\big[\sigma(g_{11})^2\big]^2 = n^{-2}$, this is enough to conclude a $C/n$ bound on the magnitude of the linear term. Thus, in total, we have shown

$$|\Xi_1| \leq \frac{C}{n} + C'e^{-cn} + C''n^{1/2}e^{-c'n},$$

where we combine the different constant that appear in the various exponential additive errors throughout our work by choosing the largest magnitude scaling factor and the smallest magnitude constant in the exponent to assert the previous expression.

$\Xi_2$ **control.** The approach is similar to what we have used to control $\Xi_1$. We start with exactly the same $\mathcal{E}_1$ event definition, and as previously define

$$\widetilde{\Xi}_2 = \mathbb{E}\left[\mathbb{1}_{\mathcal{E}_1} \frac{\langle \dot{\boldsymbol{v}}_0, \boldsymbol{v}_\nu \rangle \langle \boldsymbol{v}_\nu, \dot{\boldsymbol{v}}_\nu \rangle \|\boldsymbol{v}_0\|_2^2}{\|\boldsymbol{v}_0\|_2^3 \|\boldsymbol{v}_\nu\|_2^3}\right],$$

and then calculating

$$
\begin{aligned}
|\widetilde{\Xi}_2 - \Xi_2| &= \left|\mathbb{E}\left[\mathbb{1}_{\mathcal{E}_m \setminus \mathcal{E}} \frac{\langle \dot{\boldsymbol{v}}_0, \boldsymbol{v}_\nu \rangle \langle \boldsymbol{v}_\nu, \dot{\boldsymbol{v}}_\nu \rangle \|\boldsymbol{v}_0\|_2^2}{\|\boldsymbol{v}_0\|_2^3 \|\boldsymbol{v}_\nu\|_2^3}\right]\right| \\
&\leq 2^6 \mathbb{E}\big[\mathbb{1}_{\mathcal{E}_m \setminus \mathcal{E}} |\langle \dot{\boldsymbol{v}}_0, \boldsymbol{v}_\nu \rangle \langle \boldsymbol{v}_\nu, \dot{\boldsymbol{v}}_\nu \rangle| \|\boldsymbol{v}_0\|_2^2\big] \\
&\leq 2^6 \mathbb{E}\big[\mathbb{1}_{\mathcal{E}^c} |\langle \dot{\boldsymbol{v}}_0, \boldsymbol{v}_\nu \rangle \langle \boldsymbol{v}_\nu, \dot{\boldsymbol{v}}_\nu \rangle| \|\boldsymbol{v}_0\|_2^2\big] \\
&\leq 2^6 \mathbb{E}[\mathbb{1}_{\mathcal{E}^c}]^{1/2} \mathbb{E}\big[\langle \dot{\boldsymbol{v}}_0, \boldsymbol{v}_\nu \rangle^4\big]^{1/4} \mathbb{E}\big[\langle \boldsymbol{v}_\nu, \dot{\boldsymbol{v}}_\nu \rangle^8\big]^{1/8} \mathbb{E}\big[\|\boldsymbol{v}_0\|_2^8\big]^{1/8} \\
&\leq 2^6 \mathbb{E}[\mathbb{1}_{\mathcal{E}^c}]^{1/2} \mathbb{E}\big[\|\dot{\boldsymbol{v}}_0\|_2^8\big]^{1/8} \mathbb{E}\big[\|\boldsymbol{v}_\nu\|_2^8\big]^{1/8} \mathbb{E}\big[\|\boldsymbol{v}_\nu\|_2^{16}\big]^{1/16} \mathbb{E}\big[\|\dot{\boldsymbol{v}}_\nu\|_2^{16}\big]^{1/16} \mathbb{E}\big[\|\boldsymbol{v}_0\|_2^8\big]^{1/8} \\
&\leq Ce^{-cn} + C'n^{1/2}e^{-c'n},
\end{aligned}
$$

using the same ideas as in the previous section, plus several applications of the Schwarz inequality and a final application of Lemma E.29. We can therefore pass to $\widetilde{\Xi}_2$ with a small additive error. Next, we Taylor expand in the same way as previously, except that larger powers in the denominator force the constant in our residual bound to be $3 \cdot 2^{27}$, and the event $\mathcal{E}_1$ now gives us a bound $|\langle \dot{\boldsymbol{v}}_0, \boldsymbol{v}_\nu \rangle \langle \boldsymbol{v}_\nu, \dot{\boldsymbol{v}}_\nu \rangle \|\boldsymbol{v}_0\|_2^2| \leq 2^6$ on the numerator, which we add and subtract as before to exploit nonnegativity. We get

$$\widetilde{\Xi}_2 \leq \mathbb{E}\left[\mathbb{1}_{\mathcal{E}_1} \langle \dot{\boldsymbol{v}}_0, \boldsymbol{v}_\nu \rangle \langle \boldsymbol{v}_\nu, \dot{\boldsymbol{v}}_\nu \rangle \|\boldsymbol{v}_0\|_2^2 \left(\frac{1}{2}\left(3 - \|\boldsymbol{v}_0\|_2^6 \|\boldsymbol{v}_\nu\|_2^6\right) + (2^6 + 1)C\left(\|\boldsymbol{v}_0\|_2^6 \|\boldsymbol{v}_\nu\|_2^6 - 1\right)^2\right)\right];$$

$$\widetilde{\Xi}_2 \geq \mathbb{E}\left[\mathbb{1}_{\mathcal{E}_1} \langle \dot{\boldsymbol{v}}_0, \boldsymbol{v}_\nu \rangle \langle \boldsymbol{v}_\nu, \dot{\boldsymbol{v}}_\nu \rangle \|\boldsymbol{v}_0\|_2^2 \left(\frac{1}{2}\left(3 - \|\boldsymbol{v}_0\|_2^6 \|\boldsymbol{v}_\nu\|_2^6\right) - 2^6 C\left(\|\boldsymbol{v}_0\|_2^6 \|\boldsymbol{v}_\nu\|_2^6 - 1\right)^2\right)\right],$$

with $C = 3 \cdot 2^{27}$. Proceeding to control the quadratic term, we have

$$
\begin{aligned}
\bigg|\mathbb{E}\Big[\mathbb{1}_{\mathcal{E}_1} &\langle \dot{\boldsymbol{v}}_0, \boldsymbol{v}_\nu \rangle \langle \boldsymbol{v}_\nu, \dot{\boldsymbol{v}}_\nu \rangle \|\boldsymbol{v}_0\|_2^2 \left(\|\boldsymbol{v}_0\|_2^6 \|\boldsymbol{v}_\nu\|_2^6 - 1\right)^2\Big]\bigg| \\
&\leq 4^3 \mathbb{E}\Big[\left(\|\boldsymbol{v}_0\|_2^6 \|\boldsymbol{v}_\nu\|_2^6 - 1\right)^2\Big] \\
&= 2^6 \mathbb{E}\big[\|\boldsymbol{v}_0\|_2^{12} \|\boldsymbol{v}_\nu\|_2^{12} - 2\|\boldsymbol{v}_0\|_2^6 \|\boldsymbol{v}_\nu\|_2^6 + 1\big] \\
&\leq 2^6 \left(1 - 2\mathbb{E}[\|\boldsymbol{v}_0\|_2^6 \|\boldsymbol{v}_\nu\|_2^6] + \mathbb{E}\big[\|\boldsymbol{v}_0\|_2^{24}\big]\right) \\
&\leq 2^6 \left(1 - 2(1 - (Cn^{-1} + C'e^{-cn}))^6 + (1 + C''n^{-1})\right) \\
&\leq Cn^{-1} + \sum_{k=1}^{3} \binom{6}{2k-1} \left(C'n^{-1} + C''e^{-cn}\right)^{2k-1} \\
&\leq Cn^{-1} + C' \sum_{k=1}^{3} \sum_{j=0}^{2k-1} n^{-(2k-1-j)} e^{-cnj} \\
&\leq Cn^{-1} + C'e^{-cn}.
\end{aligned}
$$

The justifications for the first four lines are identical to those of the previous section. In the last three lines, we use the binomial theorem twice to expand the sixth power term, and we assert the final line

by the fact that $k > 0$, so that each term in the sum corresponding to a $j = 0$ has a positive inverse power of $n$ attached, and when $j = 2k - 1$ we pick up an exponential factor. Moving on to the linear term, as in the previous section we start by dropping the indicator. We control the residual as follows:

$$\left| \mathbb{E}\big[\mathbb{1}_{\mathcal{E}_1^c}\langle \dot{\boldsymbol{v}}_0, \boldsymbol{v}_\nu \rangle \langle \boldsymbol{v}_\nu, \dot{\boldsymbol{v}}_\nu \rangle \|\boldsymbol{v}_0\|_2^2 \left( \|\boldsymbol{v}_0\|_2^6 \|\boldsymbol{v}_\nu\|_2^6 - 3 \right) \big] \right|$$

$$\leq \mathbb{E}\big[\mathbb{1}_{\mathcal{E}_1^c}\big]^{1/2} \mathbb{E}\Big[ \|\dot{\boldsymbol{v}}_0\|_2^2 \|\dot{\boldsymbol{v}}_\nu\|_2^2 \|\boldsymbol{v}_0\|_2^2 \|\boldsymbol{v}_\nu\|_2^4 \left( \|\boldsymbol{v}_0\|_2^6 \|\boldsymbol{v}_\nu\|_2^6 - 3 \right)^2 \Big]^{1/2}$$

$$\leq \mathbb{E}\big[\mathbb{1}_{\mathcal{E}_1^c}\big]^{1/2} \mathbb{E}\big[\|\dot{\boldsymbol{v}}_0\|_2^2 \|\dot{\boldsymbol{v}}_\nu\|_2^2 \|\boldsymbol{v}_0\|_2^{14} \|\boldsymbol{v}_\nu\|_2^{16} + 3\|\dot{\boldsymbol{v}}_0\|_2^2 \|\dot{\boldsymbol{v}}_\nu\|_2^2 \|\boldsymbol{v}_0\|_2^2 \|\boldsymbol{v}_\nu\|_2^4 \big]^{1/2}$$

$$\leq \mathbb{E}\big[\mathbb{1}_{\mathcal{E}_1^c}\big]^{1/2} \left( \mathbb{E}\big[\|\dot{\boldsymbol{v}}_0\|_2^2 \|\dot{\boldsymbol{v}}_\nu\|_2^2 \|\boldsymbol{v}_0\|_2^{14} \|\boldsymbol{v}_\nu\|_2^{16} \big]^{1/2} + 3\mathbb{E}\big[\|\dot{\boldsymbol{v}}_0\|_2^2 \|\dot{\boldsymbol{v}}_\nu\|_2^2 \|\boldsymbol{v}_0\|_2^2 \|\boldsymbol{v}_\nu\|_2^4 \big]^{1/2} \right)$$

$$\leq C e^{-cn} + C' n^{1/2} e^{-c'n}.$$

The justifications are almost the same as the previous section, although we have compressed some steps into fewer lines here and we have omitted the final simplifications which follow from applying the Schwarz inequality to each of the two expectations in the second-to-last line 3 times and then applying Lemma E.29. Dropping the indicator and distributing now gives:

$$\mathbb{E}\big[\langle \dot{\boldsymbol{v}}_0, \boldsymbol{v}_\nu \rangle \langle \boldsymbol{v}_\nu, \dot{\boldsymbol{v}}_\nu \rangle \|\boldsymbol{v}_0\|_2^2 \left( \|\boldsymbol{v}_0\|_2^6 \|\boldsymbol{v}_\nu\|_2^6 - 3 \right)\big] = \begin{array}{l} \mathbb{E}\big[\langle \dot{\boldsymbol{v}}_0, \boldsymbol{v}_\nu \rangle \langle \boldsymbol{v}_\nu, \dot{\boldsymbol{v}}_\nu \rangle \|\boldsymbol{v}_0\|_2^2 \|\boldsymbol{v}_0\|_2^6 \|\boldsymbol{v}_\nu\|_2^6\big] \\ -3\mathbb{E}\big[\langle \dot{\boldsymbol{v}}_0, \boldsymbol{v}_\nu \rangle \langle \boldsymbol{v}_\nu, \dot{\boldsymbol{v}}_\nu \rangle \|\boldsymbol{v}_0\|_2^2\big]; \end{array}$$

to apply Lemma E.30, we check the two new coordinate functions that appear in this linear term: for $\langle \dot{\boldsymbol{v}}_0, \boldsymbol{v}_\nu \rangle$, we have

$$\mathbb{E}\big[|\dot{\sigma}(g_{1i}) g_{2i} \sigma(g_{1i} \cos \nu + g_{2i} \sin \nu)|^k\big] \leq \mathbb{E}\big[\dot{\sigma}(g_{1i}) g_{2i}^{2k}\big]^{1/2} \mathbb{E}\big[\sigma(g_{1i})^{2k}\big]^{1/2}, \qquad (\text{E.34})$$

and for $\langle \dot{\boldsymbol{v}}_\nu, \boldsymbol{v}_\nu \rangle$, we have likewise

$$\mathbb{E}\big[|\sigma(g_{1i} \cos \nu + g_{2i} \sin \nu)(g_{2i} \cos \nu - g_{1i} \sin \nu)|^k\big] \leq \mathbb{E}\big[\dot{\sigma}(g_{1i}) g_{2i}^{2k}\big]^{1/2} \mathbb{E}\big[\sigma(g_{1i})^{2k}\big]^{1/2}, \quad (\text{E.35})$$

both by the Schwarz inequality and rotational invariance. As before, an appeal to Lemmas G.11 and E.17 implies that these two coordinate functions satisfy the hypotheses of Lemma E.30, so we have a bound

$$\left| \mathbb{E}\big[\langle \dot{\boldsymbol{v}}_0, \boldsymbol{v}_\nu \rangle \langle \boldsymbol{v}_\nu, \dot{\boldsymbol{v}}_\nu \rangle \|\boldsymbol{v}_0\|_2^2 \left( \|\boldsymbol{v}_0\|_2^6 \|\boldsymbol{v}_\nu\|_2^6 - 3 \right)\big] - n^9 \mathbb{E}[(\dot{\boldsymbol{v}}_0)_1 (\boldsymbol{v}_\nu)_1]\mathbb{E}[(\dot{\boldsymbol{v}}_\nu)_1 (\boldsymbol{v}_\nu)_1]\mathbb{E}\big[\sigma(w_{11})^2\big]^7 \right.$$

$$\left. + 3n^3 \mathbb{E}[(\dot{\boldsymbol{v}}_0)_1 (\dot{\boldsymbol{v}}_\nu)_1]\mathbb{E}[(\dot{\boldsymbol{v}}_\nu)_1 (\boldsymbol{v}_\nu)_1]\mathbb{E}\big[\sigma(w_{11})^2\big] \right| \leq \frac{C}{n}.$$

Noticing that

$$\mathbb{E}[\langle \boldsymbol{v}_\nu, \dot{\boldsymbol{v}}_\nu \rangle] = -\mathbb{E}[\langle \boldsymbol{v}_0, \dot{\boldsymbol{v}}_0 \rangle] = -\mathbb{E}[\langle \sigma(\boldsymbol{g}_1), \boldsymbol{g}_2 \rangle] = 0,$$

by rotational invariance and independence, we conclude by identically-distributedness of the coordinates of $\boldsymbol{v}_\nu$ and $\dot{\boldsymbol{v}}_\nu$

$$n^9 \mathbb{E}[(\dot{\boldsymbol{v}}_0)_1 (\boldsymbol{v}_\nu)_1]\mathbb{E}[(\dot{\boldsymbol{v}}_\nu)_1 (\boldsymbol{v}_\nu)_1]\mathbb{E}\big[\sigma(g_{11})^2\big]^7 - 3n^3 \mathbb{E}[(\dot{\boldsymbol{v}}_0)_1 (\dot{\boldsymbol{v}}_\nu)_1]\mathbb{E}[(\dot{\boldsymbol{v}}_\nu)_1 (\boldsymbol{v}_\nu)_1]\mathbb{E}\big[\sigma(g_{11})^2\big] = 0,$$

which establishes the desired control on $\Xi_2$. Thus, in total, we have shown

$$|\Xi_2| \leq \frac{C}{n} + C' e^{-cn} + C'' n^{1/2} e^{-c'n},$$

where we combine the different constant that appear in the various exponential additive errors throughout our work by choosing the largest magnitude scaling factor and the smallest magnitude constant in the exponent to assert the previous expression.

$\Xi_3$ **control.** The argument for control of this term is very similar to the previous section, since the degrees of the denominators now match. We start by defining

$$\widetilde{\Xi}_3 = \mathbb{E}\left[\mathbb{1}_{\mathcal{E}_1} \frac{\langle \dot{\boldsymbol{v}}_0, \boldsymbol{v}_0 \rangle \langle \boldsymbol{v}_0, \boldsymbol{v}_\nu \rangle \langle \boldsymbol{v}_\nu, \dot{\boldsymbol{v}}_\nu \rangle}{\|\boldsymbol{v}_0\|_2^3 \|\boldsymbol{v}_\nu\|_2^3}\right],$$

with the same $\mathcal{E}_1$ event as previously, and then calculating

$$
\begin{aligned}
|\widetilde{\Xi}_3 - \Xi_3| &= \left| \mathbb{E}\left[ \mathbb{1}_{\mathcal{E}_m \setminus \mathcal{E}} \frac{\langle \dot{\boldsymbol{v}}_0, \boldsymbol{v}_0 \rangle \langle \boldsymbol{v}_\nu, \dot{\boldsymbol{v}}_\nu \rangle \langle \boldsymbol{v}_0, \boldsymbol{v}_\nu \rangle}{\|\boldsymbol{v}_0\|_2^3 \|\boldsymbol{v}_\nu\|_2^3} \right] \right| \\
&\le 2^6 \mathbb{E}\left[ \mathbb{1}_{\mathcal{E}_m \setminus \mathcal{E}} |\langle \dot{\boldsymbol{v}}_0, \boldsymbol{v}_0 \rangle \langle \boldsymbol{v}_\nu, \dot{\boldsymbol{v}}_\nu \rangle \langle \boldsymbol{v}_0, \boldsymbol{v}_\nu \rangle| \right] \\
&\le 2^6 \mathbb{E}[\mathbb{1}_{\mathcal{E}^c} |\langle \dot{\boldsymbol{v}}_0, \boldsymbol{v}_0 \rangle \langle \boldsymbol{v}_\nu, \dot{\boldsymbol{v}}_\nu \rangle \langle \boldsymbol{v}_0, \boldsymbol{v}_\nu \rangle|] \\
&\le 2^6 \mathbb{E}[\mathbb{1}_{\mathcal{E}^c}]^{1/2} \mathbb{E}\left[ \langle \dot{\boldsymbol{v}}_0, \boldsymbol{v}_0 \rangle^4 \right]^{1/4} \mathbb{E}\left[ \langle \boldsymbol{v}_\nu, \dot{\boldsymbol{v}}_\nu \rangle^8 \right]^{1/8} \mathbb{E}\left[ \langle \boldsymbol{v}_0, \boldsymbol{v}_\nu \rangle^8 \right]^{1/8} \\
&\le 2^6 \mathbb{E}[\mathbb{1}_{\mathcal{E}^c}]^{1/2} \mathbb{E}\left[ \|\dot{\boldsymbol{v}}_0\|_2^8 \right]^{1/8} \mathbb{E}\left[ \|\boldsymbol{v}_0\|_2^8 \right]^{1/8} \mathbb{E}\left[ \|\dot{\boldsymbol{v}}_\nu\|_2^{16} \right]^{1/16} \mathbb{E}\left[ \|\boldsymbol{v}_0\|_2^{16} \right]^{1/16} \mathbb{E}\left[ \|\boldsymbol{v}_\nu\|_2^{16} \right]^{1/8} \\
&\le C n^{1/2} e^{-cn} + C e^{-c'n},
\end{aligned}
$$

using the same ideas as in the previous section. We can therefore pass to $\widetilde{\Xi}_3$ with an exponentially small error. Next, we Taylor expand in the same way as previously, obtaining

$$
\widetilde{\Xi}_3 \le \mathbb{E}\left[ \mathbb{1}_{\mathcal{E}_1} \langle \boldsymbol{v}_0, \boldsymbol{v}_\nu \rangle \langle \dot{\boldsymbol{v}}_0, \boldsymbol{v}_0 \rangle \langle \boldsymbol{v}_\nu, \dot{\boldsymbol{v}}_\nu \rangle \left( \frac{1}{2}\left(3 - \|\boldsymbol{v}_0\|_2^6 \|\boldsymbol{v}_\nu\|_2^6\right) + (4^3+1)C\left(\|\boldsymbol{v}_0\|_2^6 \|\boldsymbol{v}_\nu\|_2^6 - 1\right)^2 \right) \right];
$$

$$
\widetilde{\Xi}_3 \ge \mathbb{E}\left[ \mathbb{1}_{\mathcal{E}_1} \langle \boldsymbol{v}_0, \boldsymbol{v}_\nu \rangle \langle \dot{\boldsymbol{v}}_0, \boldsymbol{v}_0 \rangle \langle \boldsymbol{v}_\nu, \dot{\boldsymbol{v}}_\nu \rangle \left( \frac{1}{2}\left(3 - \|\boldsymbol{v}_0\|_2^6 \|\boldsymbol{v}_\nu\|_2^6\right) - 4^3 C\left(\|\boldsymbol{v}_0\|_2^6 \|\boldsymbol{v}_\nu\|_2^6 - 1\right)^2 \right) \right],
$$

with $C = 3 \cdot 2^{27}$. Proceeding to control the quadratic term, we notice

$$
\left| \mathbb{E}\left[ \mathbb{1}_{\mathcal{E}_1} \langle \dot{\boldsymbol{v}}_0, \boldsymbol{v}_0 \rangle \langle \boldsymbol{v}_\nu, \dot{\boldsymbol{v}}_\nu \rangle \langle \boldsymbol{v}_0, \boldsymbol{v}_\nu \rangle \left(\|\boldsymbol{v}_0\|_2^6 \|\boldsymbol{v}_\nu\|_2^6 - 1\right)^2 \right] \right| \le 4^3 \mathbb{E}\left[ \left(\|\boldsymbol{v}_0\|_2^6 \|\boldsymbol{v}_\nu\|_2^6 - 1\right)^2 \right]
$$
$$
= C n^{-1} + C' e^{-cn},
$$

since the final term was controlled in the previous section. Moving on to the linear term, as in the previous section we start by dropping the indicator. We control the residual as follows:

$$
\begin{aligned}
&\left| \mathbb{E}\left[ \mathbb{1}_{\mathcal{E}_1^c} \langle \dot{\boldsymbol{v}}_0, \boldsymbol{v}_0 \rangle \langle \boldsymbol{v}_\nu, \dot{\boldsymbol{v}}_\nu \rangle \langle \boldsymbol{v}_0, \boldsymbol{v}_\nu \rangle \left(\|\boldsymbol{v}_0\|_2^6 \|\boldsymbol{v}_\nu\|_2^6 - 3\right) \right] \right| \\
&\le \mathbb{E}\left[\mathbb{1}_{\mathcal{E}_1^c}\right]^{1/2} \mathbb{E}\left[ \|\dot{\boldsymbol{v}}_0\|_2^2 \|\dot{\boldsymbol{v}}_\nu\|_2^2 \|\boldsymbol{v}_0\|_2^4 \|\boldsymbol{v}_\nu\|_2^4 \left(\|\boldsymbol{v}_0\|_2^6 \|\boldsymbol{v}_\nu\|_2^6 - 3\right)^2 \right]^{1/2} \\
&\le \mathbb{E}\left[\mathbb{1}_{\mathcal{E}_1^c}\right]^{1/2} \mathbb{E}\left[ \|\dot{\boldsymbol{v}}_0\|_2^2 \|\dot{\boldsymbol{v}}_\nu\|_2^2 \|\boldsymbol{v}_0\|_2^{16} \|\boldsymbol{v}_\nu\|_2^{16} + 3\|\dot{\boldsymbol{v}}_0\|_2^2 \|\dot{\boldsymbol{v}}_\nu\|_2^2 \|\boldsymbol{v}_0\|_2^4 \|\boldsymbol{v}_\nu\|_2^4 \right]^{1/2} \\
&\le \mathbb{E}\left[\mathbb{1}_{\mathcal{E}_1^c}\right]^{1/2} \left( \mathbb{E}\left[ \|\dot{\boldsymbol{v}}_0\|_2^2 \|\dot{\boldsymbol{v}}_\nu\|_2^2 \|\boldsymbol{v}_0\|_2^{16} \|\boldsymbol{v}_\nu\|_2^{16} \right]^{1/2} + 3\mathbb{E}\left[ \|\dot{\boldsymbol{v}}_0\|_2^2 \|\dot{\boldsymbol{v}}_\nu\|_2^2 \|\boldsymbol{v}_0\|_2^4 \|\boldsymbol{v}_\nu\|_2^4 \right]^{1/2} \right) \\
&\le C e^{-cn} + C' n^{1/2} e^{-c'n},
\end{aligned}
$$

by the same argument as in the previous section. Dropping the indicator and distributing now gives:

$$
\mathbb{E}\left[ \langle \dot{\boldsymbol{v}}_0, \boldsymbol{v}_0 \rangle \langle \boldsymbol{v}_\nu, \dot{\boldsymbol{v}}_\nu \rangle \langle \boldsymbol{v}_0, \boldsymbol{v}_\nu \rangle \left(\|\boldsymbol{v}_0\|_2^6 \|\boldsymbol{v}_\nu\|_2^6 - 3\right) \right] = \begin{aligned} &\mathbb{E}\left[ \langle \dot{\boldsymbol{v}}_0, \boldsymbol{v}_0 \rangle \langle \boldsymbol{v}_\nu, \dot{\boldsymbol{v}}_\nu \rangle \langle \boldsymbol{v}_0, \boldsymbol{v}_\nu \rangle \|\boldsymbol{v}_0\|_2^6 \|\boldsymbol{v}_\nu\|_2^6 \right] \\ &- 3\mathbb{E}\left[ \langle \dot{\boldsymbol{v}}_0, \boldsymbol{v}_0 \rangle \langle \boldsymbol{v}_\nu, \dot{\boldsymbol{v}}_\nu \rangle \langle \boldsymbol{v}_0, \boldsymbol{v}_\nu \rangle \right]; \end{aligned}
$$

to apply Lemma E.30, we check the one new coordinate function that appears in this linear term: for $\langle \boldsymbol{v}_0, \boldsymbol{v}_\nu \rangle$, we have

$$
\mathbb{E}\left[ |\sigma(g_{1i})\sigma(g_{1i}\cos\nu + g_{2i}\sin\nu)|^k \right] \le \mathbb{E}\left[ \sigma(g_{1i})^{2k} \right], \tag{E.36}
$$

by the Schwarz inequality and rotational invariance. As before, an appeal to Lemmas G.11 and E.17 implies that this coordinate function satisfies the hypotheses of Lemma E.30, so we have a bound

$$
\left| \begin{aligned} &\mathbb{E}\left[ \langle \dot{\boldsymbol{v}}_0, \boldsymbol{v}_0 \rangle \langle \boldsymbol{v}_\nu, \dot{\boldsymbol{v}}_\nu \rangle \langle \boldsymbol{v}_0, \boldsymbol{v}_\nu \rangle \left(\|\boldsymbol{v}_0\|_2^6 \|\boldsymbol{v}_\nu\|_2^6 - 3\right) \right] \\ &- n^9 \mathbb{E}[(\dot{\boldsymbol{v}}_0)_1 (\boldsymbol{v}_0)_1] \mathbb{E}[(\dot{\boldsymbol{v}}_\nu)_1 (\boldsymbol{v}_\nu)_1] \mathbb{E}[(\boldsymbol{v}_\nu)_1 (\boldsymbol{v}_0)_1] \mathbb{E}\left[ \sigma(g_{11})^2 \right]^6 \\ &+ 3n^3 \mathbb{E}[(\dot{\boldsymbol{v}}_0)_1 (\boldsymbol{v}_0)_1] \mathbb{E}[(\dot{\boldsymbol{v}}_\nu)_1 (\boldsymbol{v}_\nu)_1] \mathbb{E}[(\boldsymbol{v}_\nu)_1 (\boldsymbol{v}_0)_1] \end{aligned} \right| \lesssim n^{-1}.
$$

As in the previous section, using that $\mathbb{E}[\langle \boldsymbol{v}_\nu, \dot{\boldsymbol{v}}_\nu \rangle] = 0$ then allows us to conclude the desired control on $\Xi_3$. Thus, in total, we have shown

$$
|\Xi_3| \le \frac{C}{n} + C' e^{-cn} + C'' n^{1/2} e^{-c'n},
$$

where we combine the different constant that appear in the various exponential additive errors throughout our work by choosing the largest magnitude scaling factor and the smallest magnitude constant in the exponent to assert the previous expression.

To wrap up, we take the largest of the scaling constants in the estimates we have derived, and the smallest of the constants-in-the-exponent that we have derived, in order to assert

$$|h''(\nu)| \leq \frac{C}{n} + C'n^{1/2}e^{-cn}.$$

Matching constants in the exponent and choosing $n$ larger than an absolute constant multiple of $\log n$, it follows

$$|h(\nu)| \leq Ce^{-cn} + C'\frac{\nu^2}{n},$$

which was to be proved. □

### E.3.4    GENERAL PROPERTIES

**Lemma E.16.** *Consider the event*

$$\mathcal{E}_{c,m} = \bigcap_{\substack{S \subset [n] \\ |S|=m}} \bigcap_{\nu \in [0,2\pi]} \left\{ (\boldsymbol{g}_1, \boldsymbol{g}_2) \,\big|\, c \leq \|\boldsymbol{I}_{S^c}\boldsymbol{v}_\nu(\boldsymbol{g}_1, \boldsymbol{g}_2)\|_2 \leq c^{-1} \right\}.$$

*Suppose $n \geq \max\{2m, m+20\}$. Then we have the following properties:*

1. *$\mu(\mathcal{E}_{c,m}^{\mathsf{c}}) \leq Cn^m e^{-c'n}$;*

2. *We have $\mathcal{E}_{c,m} = \mathcal{E}_{c,m}\boldsymbol{Q}$ for every $\boldsymbol{Q} \in \mathrm{O}(2)$, so that in particular $\mathbb{1}_{\mathcal{E}_{c,m}}(\boldsymbol{G}\boldsymbol{Q}) = \mathbb{1}_{\mathcal{E}_{c,m}}(\boldsymbol{G})$.*

*Above, $\mathrm{O}(n)$ denotes the set of $n \times n$ orthogonal matrices.*

*Proof.* We will show the second property first. For each $c > 0$, if $\boldsymbol{Q} \in \mathrm{O}(2)$, notice that

$$\mathcal{E}_{c,m}\boldsymbol{Q} = \bigcap_{\substack{S \subset [n] \\ |S|=m}} \bigcap_{\nu \in [0,2\pi]} \left\{ \boldsymbol{G}\boldsymbol{Q} \,\bigg|\, c < \left\| \boldsymbol{I}_{S^c}\sigma\left(\boldsymbol{G}\begin{bmatrix}\cos\nu \\ \sin\nu\end{bmatrix}\right) \right\|_2 < c^{-1} \right\}$$

$$= \bigcap_{\substack{S \subset [n] \\ |S|=m}} \bigcap_{\boldsymbol{u} \in \mathbb{S}^1} \left\{ \boldsymbol{G} \,\big|\, c < \|\boldsymbol{I}_{S^c}\sigma\left(\boldsymbol{G}\boldsymbol{Q}^*\boldsymbol{u}\right)\|_2 < c^{-1} \right\}$$

$$= \mathcal{E}_{c,m},$$

since the vector $[\cos\nu, \sin\nu]^* \in \mathbb{S}^1$, and $\mathrm{O}(2)$ acts transitively on $\mathbb{S}^1$. This proves the second property when $c > 0$; the result for $c = 0$ is obtained by applying the preceding argument to each set in the infinite union defining the $c = 0, m$ event.

For the measure bound, we observe that $\mathcal{E}_{c,m} \subset \mathcal{E}_{c',m}$ if $c \geq c'$, so it suffices to bound the measure of the complement for the particular choice $c = \frac{1}{2}$. We start by controlling pointwise the measure of the complement of the event

$$\mathcal{E}_{0.6,m,\boldsymbol{u}} = \bigcap_{\substack{S \subset [n] \\ |S|=m}} \left\{ \boldsymbol{G} \,|\, 0.6 < \|\boldsymbol{I}_{S^c}\sigma\left(\boldsymbol{G}\boldsymbol{u}\right)\|_2 < 5/3 \right\}$$

for each $\boldsymbol{u} \in \mathbb{S}^1$, then uniformize over the one-dimensional manifold $\mathbb{S}^1$; we need to begin with $c = 0.6$ instead of $c = \frac{1}{2}$ to survive some loosening of the bounds when we uniformize. We have

$$\mathcal{E}_{0.6,m,\boldsymbol{u}}^{\mathsf{c}} = \bigcup_{\substack{S \subset [n] \\ |S|=m}} \left\{ \boldsymbol{G} \,|\, \|\boldsymbol{I}_{S^c}\sigma\left(\boldsymbol{G}\boldsymbol{u}\right)\|_2 \leq 0.6 \right\} \cup \left\{ \boldsymbol{G} \,|\, \|\boldsymbol{I}_{S^c}\sigma\left(\boldsymbol{G}\boldsymbol{u}\right)\|_2 \geq 5/3 \right\},$$

so that a union bound implies

$$\mu\left(\mathcal{E}_{0.6,m,\boldsymbol{u}}^{\mathsf{c}}\right) \leq \sum_{\substack{S \subset [n] \\ |S|=m}} \mathbb{P}[\|\boldsymbol{I}_{S^c}\sigma\left(\boldsymbol{G}\boldsymbol{u}\right)\|_2 \leq 0.6] + \mathbb{P}[\|\boldsymbol{I}_{S^c}\sigma\left(\boldsymbol{G}\boldsymbol{u}\right)\|_2 \geq 5/3]$$

$$\leq \binom{n}{m}\left(\mathbb{P}[\|\boldsymbol{I}_{[m]^c}\sigma\left(\boldsymbol{g}_1\right)\|_2 \leq 0.6] + \mathbb{P}[\|\boldsymbol{I}_{[m]^c}\sigma\left(\boldsymbol{g}_1\right)\|_2 \geq 5/3]\right), \qquad \text{(E.37)}$$

where the final inequality follows from right-rotational invariance of $\mu$ and identically-distributedness of the coordinates of $\boldsymbol{g}_1$. Let $\tilde{\boldsymbol{g}} \in \mathbb{R}^{n-m}$ be distributed as $\mathcal{N}(\boldsymbol{0}, (2/n)\boldsymbol{I})$, so that $\sigma(\tilde{\boldsymbol{g}})$ has the same distribution as $\boldsymbol{I}_{[m]^c}\sigma(\boldsymbol{g}_1)$ By Gauss-Lipschitz concentration (Boucheron et al., 2013, Theorem 5.6), we have

$$\mathbb{P}[\|\sigma(\tilde{\boldsymbol{g}})\|_2 \geq \mathbb{E}[\|\sigma(\tilde{\boldsymbol{g}})\|_2] + t] \leq e^{-cnt^2}, \qquad \mathbb{P}[\|\sigma(\tilde{\boldsymbol{g}})\|_2 \leq \mathbb{E}[\|\sigma(\tilde{\boldsymbol{g}})\|_2] - t] \leq e^{-cnt^2},$$

since $\sigma$ is 1-Lipschitz and nonnegative homogeneous. After rescaling, we apply Lemma E.19 to get

$$\sqrt{1 - \frac{m}{n}} - \frac{2}{\sqrt{n}\sqrt{n-m}} \leq \mathbb{E}[\|\sigma(\tilde{\boldsymbol{g}})\|_2] \leq \sqrt{1 - \frac{m}{n}} \leq 1$$

Plugging these estimates into the Gauss-Lipschitz bounds gives

$$\mathbb{P}[\|\sigma(\tilde{\boldsymbol{g}})\|_2 \geq 1 + t] \leq e^{-cnt^2}, \qquad \mathbb{P}\left[\|\sigma(\tilde{\boldsymbol{g}})\|_2 \leq \sqrt{1 - \frac{m}{n}} - \frac{2}{\sqrt{n}\sqrt{n-m}} - t\right] \leq e^{-cnt^2}.$$

Putting $t = 2/3$ in the upper tail bound gives the control we need for one half of (E.37). For the lower tail, we note that the assumption $n \geq \max\{2m, m+20\}$ yields the estimates

$$\sqrt{1 - \frac{m}{n}} \geq \frac{1}{\sqrt{2}}, \qquad \frac{2}{\sqrt{n}\sqrt{n-m}} \leq \frac{2}{n-m} \leq \frac{1}{10},$$

so that

$$\sqrt{1 - \frac{m}{n}} - \frac{2}{\sqrt{n}\sqrt{n-m}} - t \geq \frac{1}{\sqrt{2}} - \frac{1}{10} - t,$$

and one checks numerically that $2^{-1/2} - (1/10) > 0.6$. Putting therefore $t = 2^{-1/2} - (1/10) - 0.6$ in the lower tail bound yields

$$\mathbb{P}[\|\sigma(\tilde{\boldsymbol{g}})\|_2 \leq 0.6] \leq e^{-cn}.$$

Plugging these results into (E.37) gives the pointwise measure bound

$$\mu\left(\mathcal{E}_{0.6,m,\boldsymbol{u}}^{\mathsf{c}}\right) \leq 2\binom{n}{m}e^{-cn}$$

for some constant $c > 0$.

For uniformization, fix $S \subset [n]$ with $|S| = m$ and consider the function $f_S : \mathbb{R}^2 \to \mathbb{R}$ defined by

$$f_S(\boldsymbol{u}) = \|\boldsymbol{I}_{S^c}\sigma(\boldsymbol{G}\boldsymbol{u})\|_2.$$

By Gauss-Lipschitz concentration, we have

$$\mathbb{P}[\|\boldsymbol{G}\| > \mathbb{E}[\|\boldsymbol{G}\|] + t] \leq e^{-cnt^2},$$

and by (Rudelson & Vershynin, 2011, Theorem 2.6), we have

$$\mathbb{E}[\|\boldsymbol{G}\|] \leq \sqrt{2} + \frac{2}{\sqrt{n}} \leq 4.$$

Let $\mathcal{E} = \{\|\boldsymbol{G}\| \leq 5\}$; then it follows that $\mu(\mathcal{E}) \geq 1 - e^{-cn}$. On $\mathcal{E}$, for *every* $S$, we have that $f_S$ is a 5-Lipschitz function of $\boldsymbol{u}$. Let $T_\varepsilon \subset \mathbb{S}^1$ be a family of sets with the property that $\boldsymbol{u} \in \mathbb{S}^1$ implies that there is $\boldsymbol{u}' \in T_\varepsilon$ such that $\|\boldsymbol{u}' - \boldsymbol{u}\|_2 \leq \varepsilon$ for each $\varepsilon > 0$; by standard results (Vershynin, 2018, Corollary 4.2.13), $T_\varepsilon$ exists and we have $|T_\varepsilon| \leq (1 + 2\varepsilon^{-1})^2$. Define

$$\mathcal{E}_{0.6,m,\varepsilon} = \bigcap_{\boldsymbol{u} \in T_\varepsilon} \mathcal{E}_{0.6,m,\boldsymbol{u}}.$$

Then a union bound together with our pointwise concentration result gives

$$\mu\left(\mathcal{E}_{0.6,m,\varepsilon}^{\mathsf{c}}\right) \leq 2\binom{n}{m}\left(1 + \frac{2}{\varepsilon}\right)^2 e^{-cn}.$$

On $\mathcal{E} \cap \mathcal{E}_{0.6,m,\varepsilon}$, for any $\boldsymbol{u} \in \mathbb{S}^1$ and any $S$, there is $\boldsymbol{u}' \in T_\varepsilon$ such that $|f_S(\boldsymbol{u}) - f_S(\boldsymbol{u}')| \leq 5\varepsilon$. But since on this event $0.6 \leq f_S(\boldsymbol{u}') \leq 5/3$, we conclude $0.6 - 5\varepsilon \leq f_S(\boldsymbol{u}) \leq 5/3 + 5\varepsilon$, and therefore the choice $\varepsilon = 1/50$ gives $0.5 \leq f_S(\boldsymbol{u}) \leq 2$. This implies

$$\mathcal{E} \cap \mathcal{E}_{0.6,m,1/50} \subset \mathcal{E}_{0.5,m}.$$

Thus, by a union bound and our previous results, we have

$$
\begin{aligned}
\mu\left(\mathcal{E}_{0.5,m}^{\mathsf{c}}\right) &\leq \mu\left(\mathcal{E}^{\mathsf{c}} \cup \mathcal{E}_{0.6,m,1/50}^{\mathsf{c}}\right) \\
&\leq \mu\left(\mathcal{E}_{0.6,m,1/50}^{\mathsf{c}}\right) + e^{-cn} \\
&\leq 2 \cdot 150^2 \binom{n}{m} e^{-c'n} + e^{-cn},
\end{aligned}
$$

which is the desired measure bound. □

**Lemma E.17.** *We have for each fixed $\nu \in [0, \pi]$ that:*

1. *The coordinates of $\dot{\boldsymbol{v}}_\nu$ have subgaussian moment growth*

$$
\mathbb{E}[(\boldsymbol{v}_\nu)_i^p] \leq \frac{1}{2}\left(\frac{2p}{n}\right)^{p/2};
$$

2. *The event $\{\|\dot{\boldsymbol{v}}_\nu\|_2 \leq 2\}$ has probability at least $1 - e^{-cn}$;*

3. *The event $\{\forall \nu \in [0,\pi] \, \|\dot{\boldsymbol{v}}_\nu\|_2 \leq 4\}$ has probability at least $1 - e^{-c'n}$.*

*Proof.* We have that the coordinates of $\dot{\boldsymbol{v}}_\nu$ are i.i.d., and

$$
(\dot{\boldsymbol{v}}_\nu)_i \stackrel{d}{=} \dot{\sigma}(g_{1i})g_{2i},
$$

by rotational invariance. By independence of $\boldsymbol{g}_1$ and $\boldsymbol{g}_2$, we compute

$$
\mathbb{E}[(\dot{\boldsymbol{v}}_\nu)_i^p] = \mathbb{E}[\dot{\sigma}(g_{1i})g_{2i}^p] = \frac{1}{2}\mathbb{E}[g_{2i}^p] \leq \frac{2^{p/2}}{2n^{p/2}}p^{p/2},
$$

for each $p \geq 1$; the last inequality follows from Lemma G.11. This shows that the coordinates of $\dot{\boldsymbol{v}}_\nu$ are independent subgaussian random variables with scale parameters at most $C\sqrt{2/n}$, so we have a tail bound (Vershynin, 2018, Theorem 3.1.1)

$$
\mathbb{P}\big[\|\dot{\boldsymbol{v}}_\nu\|_2 \geq 1 + t\big] \leq e^{-cnt^2},
$$

also taking into account that $\mathbb{E}\big[(\dot{\boldsymbol{v}}_\nu)_i^2\big] = 1/n$. This shows that the event $\mathcal{E}'' = \{\|\dot{\boldsymbol{v}}_\nu\|_2 \leq 2\}$ has probability at least $1 - e^{-cn}$.

For the third assertion, we use the triangle inequality to get $\|\dot{\boldsymbol{v}}_\nu\|_2 \leq \|\boldsymbol{g}_2\|_2 + \|\boldsymbol{g}_2\|_2$, which has RHS independent of $\nu$; then applying Gauss-Lipschitz concentration gives for $t \geq 0$

$$
\mathbb{P}\Big[\|\boldsymbol{g}_i\|_2 \geq \sqrt{2} + t\Big] \leq e^{-cnt^2},
$$

using that $\mathbb{E}[\|\boldsymbol{g}_i\|_2] \leq \sqrt{\mathbb{E}[\|\boldsymbol{g}_i\|_2^2]}$. Putting $t = 0.5$ in this bound and applying a union bound, we conclude that there is an event of probability at least $1 - e^{-cn}$ on which $\|\dot{\boldsymbol{v}}_\nu\|_2 \leq 4$ uniformly in $\nu$. □

**Lemma E.18.** *There exists an absolute constant $C > 0$ such that if $n \geq C$, one has*

$$
1 - \frac{C'}{n} - C''e^{-cn} \leq \mathbb{E}_{\boldsymbol{g}_1, \boldsymbol{g}_2}[\|\boldsymbol{v}_0\|_2\|\boldsymbol{v}_\nu\|_2] \leq 1,
$$

*where $c, C', C'' > 0$ are absolute constants.*

*Proof.* For the upper bound, we apply the Schwarz inequality to get

$$
\mathbb{E}[\|\boldsymbol{v}_0\|_2\|\boldsymbol{v}_\nu\|_2] \leq \mathbb{E}[\|\boldsymbol{v}_0\|_2^2]^{1/2}\mathbb{E}[\|\boldsymbol{v}_\nu\|_2^2]^{1/2} \leq 1,
$$

by rotational invariance and Lemma G.11. For the lower bound, we will truncate and linearize the product using logarithms. Let $\mathcal{E} = \mathcal{E}_{0.5,0}$; by Lemma E.16, as long as $n \geq 20$ we have $\mu(\mathcal{E}^{\mathsf{c}}) \leq Ce^{-cn}$. Define $X = \|\boldsymbol{v}_0\|_2\|\boldsymbol{v}_\nu\|_2 \mathbb{1}_\mathcal{E} + \mathbb{1}_{\mathcal{E}^{\mathsf{c}}}$, so that

$$
X(\boldsymbol{G}) = \begin{cases} \|\boldsymbol{v}_0(\boldsymbol{G})\|_2\|\boldsymbol{v}_\nu(\boldsymbol{G})\|_2 & \boldsymbol{G} \in \mathcal{E}, \\ 1 & \text{otherwise.} \end{cases}
$$

We calculate

$$\left|\mathbb{E}[\|\boldsymbol{v}_0\|_2\|\boldsymbol{v}_\nu\|_2] - \mathbb{E}[X]\right| \le \mu(\mathcal{E}^c) + \mathbb{E}[\mathbb{1}_\mathcal{E}]^{1/2}\mathbb{E}[\|\boldsymbol{v}_0\|_2^4]^{1/2}$$
$$\le Ce^{-cn} + C'e^{-c'n}(1 + C'/n)^{1/2}$$

using the triangle inequality, the Schwarz inequality, rotational invariance, and Lemmas E.16 and E.29. It follows

$$\mathbb{E}[\|\boldsymbol{v}_0\|_2\|\boldsymbol{v}_\nu\|_2] \ge \mathbb{E}[X] - C'e^{-cn},$$

so it suffices to prove the lower bound for $X$ instead. Factoring as $X = (\|\boldsymbol{v}_0\|_2\mathbb{1}_\mathcal{E} + \mathbb{1}_{\mathcal{E}^c})(\|\boldsymbol{v}_\nu\|_2\mathbb{1}_\mathcal{E} + \mathbb{1}_{\mathcal{E}^c})$, we apply concavity of $x \mapsto \log x$, Jensen's inequality, and convexity of $x \mapsto e^x$ to get

$$\mathbb{E}[X] \ge \exp\left(\mathbb{E}\big[\log\big(\|\boldsymbol{v}_0\|_2\mathbb{1}_\mathcal{E} + \mathbb{1}_{\mathcal{E}^c}\big)\big] + \mathbb{E}\big[\log\big(\|\boldsymbol{v}_\nu\|_2\mathbb{1}_\mathcal{E} + \mathbb{1}_{\mathcal{E}^c}\big)\big]\right)$$
$$\ge 1 + \mathbb{E}\big[\log\big(\|\boldsymbol{v}_0\|_2\mathbb{1}_\mathcal{E} + \mathbb{1}_{\mathcal{E}^c}\big)\big] + \mathbb{E}\big[\log\big(\|\boldsymbol{v}_\nu\|_2\mathbb{1}_\mathcal{E} + \mathbb{1}_{\mathcal{E}^c}\big)\big]$$
$$\ge 1 + 2\mathbb{E}\big[\log\big(\|\boldsymbol{v}_0\|_2\mathbb{1}_\mathcal{E} + \mathbb{1}_{\mathcal{E}^c}\big)\big]$$

where the last equality is due to rotational invariance. Now write $Y = \|\boldsymbol{v}_0\|_2\mathbb{1}_\mathcal{E} + \mathbb{1}_{\mathcal{E}^c}$, so that by the definition of $\mathcal{E}$ we have $Y \ge \frac{1}{2}$. Taylor expansion with Lagrange remainder of the logarithm about $\mathbb{E}[Y] \ge \frac{1}{2}$ gives

$$\log(Y) = \log\mathbb{E}[Y] - \frac{1}{\mathbb{E}[Y]}\left(Y - \mathbb{E}[Y]\right) - \frac{1}{2\xi(Y)^2}\left(Y - \mathbb{E}[Y]\right)^2$$

for some $\xi(Y)$ between $\mathbb{E}[Y]$ and $Y$. Using $Y \ge \frac{1}{2}$ and taking expectations on both sides, we get

$$\mathbb{E}[\log Y] \ge \log\mathbb{E}[Y] - 2\text{Var}[Y].$$

Moreover, we have

$$\left|\mathbb{E}[Y] - \mathbb{E}[\|\boldsymbol{v}_0\|_2]\right| \le Ce^{-cn} + \mathbb{E}[\mathbb{1}_{\mathcal{E}^c}\|\boldsymbol{v}_0\|_2] \le Ce^{-cn} + C'e^{-c'n},$$

by the Schwarz inequality, and this extra exponential error can be rolled into the exponential error accrued via our use of $X$. In particular, we have

$$1 - \frac{2}{n} - Ce^{-cn} \le \mathbb{E}[Y] \le 1 + Ce^{-cn},$$

by Lemma E.19. Since $n \ge 20$, if we also enforce $n \ge C_1 := c^{-1}\log(5C/2)$ we have $2/n + Ce^{-cn} \le \frac{1}{2}$; it follows by concavity of $x \mapsto \log(1 - x)$ that we have a bound

$$\log\left(1 - \frac{2}{n} - Ce^{-cn}\right) \ge -2\log(2)\left(\frac{2}{n} + Ce^{-cn}\right),$$

which has the form claimed. It remains to upper bound $\text{Var}[Y]$; using that $Y^2 = \|\boldsymbol{v}_0\|_2^2\mathbb{1}_\mathcal{E} + \mathbb{1}_{\mathcal{E}^c}$, we have

$$\text{Var}[Y] = \mathbb{E}[Y^2] - \mathbb{E}[Y]^2 \le 1 + Ce^{-cn} - \left(1 - \frac{2}{n} - Ce^{-cn}\right)^2$$
$$= Ce^{-cn} + 2\left(\frac{2}{n} + Ce^{-cn}\right) - \left(\frac{2}{n} + Ce^{-cn}\right)^2$$
$$\le \frac{4}{n} + 3Ce^{-cn},$$

which is sufficient to conclude. $\qquad\square$

**Lemma E.19.** *One has*

$$1 - \frac{2}{n} \le \mathop{\mathbb{E}}_{\boldsymbol{g}_1,\boldsymbol{g}_2}\left[\|\boldsymbol{v}_\nu\|_2\right] \le 1.$$

*Proof.* By rotational invariance, it is equivalent to characterize the expectation of $\|\sigma(\boldsymbol{g}_1)\|_2$. By the Schwarz inequality, we have

$$\mathbb{E}[\|\boldsymbol{v}_0\|_2] \le \mathbb{E}\big[\|\boldsymbol{v}_0\|_2^2\big]^{1/2} = 1,$$

by Lemma G.11. For the lower bound, we apply the Gaussian Poincaré inequality (Boucheron et al., 2013, Theorem 3.20) and the 1-Lipschitz property of $g \mapsto \|\sigma(g)\|_2$ to get

$$\frac{n}{2}\mathbb{E}\left[(\|v_0\|_2 - \mathbb{E}[\|v_0\|_2])^2\right] \leq 1,$$

so that after distributing and applying $\mathbb{E}[\|v_0\|_2^2] = 1$, we see that

$$1 - \frac{2}{n} \leq \mathbb{E}[\|v_0\|_2]^2.$$

Because $n \geq 2$, it follows

$$\mathbb{E}[\|v_0\|_2] \geq \sqrt{1 - \frac{2}{n}} \geq 1 - \frac{2}{n},$$

where the last bound holds because $1 - 2n^{-1} \leq 1$. $\qquad\square$

**Lemma E.20.** *If $0 \leq x, y \leq 1$, we have*

$$|\cos^{-1} x - \cos^{-1} y| \leq \sqrt{|x - y|}.$$

*Proof.* Let $0 \leq x, y \leq 1$, and assume to begin that $x \leq y$. We apply the fundamental theorem of calculus and knowledge of the derivative of $\cos^{-1}$ to get

$$\cos^{-1} x - \cos^{-1} y = \int_x^y \frac{1}{\sqrt{1 - t^2}}\, dt$$

The integrand is nonnegative, so $\cos^{-1} x - \cos^{-1} y \geq 0$. Writing $\sqrt{1 - t^2} = \sqrt{1 - t}\sqrt{1 + t}$ and using $x \geq 0$, we get

$$\cos^{-1} x - \cos^{-1} y \leq \int_x^y \frac{1}{\sqrt{1 - t}}\, dt$$
$$= \sqrt{1 - x} - \sqrt{1 - y}.$$

This shows that $|\cos^{-1} x - \cos^{-1} y| \leq |\sqrt{1 - x} - \sqrt{1 - y}|$ when $x \leq y$. An almost-identical argument establishes the same when $y \leq x$, via the inequalities $0 \geq \cos^{-1} x - \cos^{-1} y \geq -(\sqrt{1 - x} - \sqrt{1 - y})$. So we have shown

$$|\cos^{-1} x - \cos^{-1} y| \leq |\sqrt{1 - x} - \sqrt{1 - y}|$$

for arbitrary $0 \leq x \leq 1$ and $0 \leq y \leq 1$. Now notice

$$|\sqrt{1 - x} - \sqrt{1 - y}|^2 \leq |\sqrt{1 - x} - \sqrt{1 - y}||\sqrt{1 - x} + \sqrt{1 - y}|$$
$$\leq |(1 - x) - (1 - y)| = |x - y|,$$

which establishes $|\cos^{-1} x - \cos^{-1} y| \leq \sqrt{|x - y|}$. $\qquad\square$

### E.3.5 DIFFERENTIATION RESULTS

**Lemma E.21.** *For $a < b$, let $f : [a, b] \to \mathbb{R}$ be a continuous function that is differentiable on $(a, b)$ except at a set of isolated points in $(a, b)$, and let $c \in \mathbb{R}$. Then $\max\{f, c\}$ is differentiable except at a set of isolated points in $(a, b)$.*

*Proof.* Let $A \subset (a, b)$ denote the set of points of differentiability of $f$, and let $B \subset (a, b)$ denote the set of points of nondifferentiability of $\max\{f, c\}$. Because finite unions of isolated sets of points in $(a, b)$ are isolated in $(a, b)$, it suffices to consider only points $x \in A$.

Fix $x \in A$, and consider the case $f(x) \neq c$. Then because $f$ is continuous, there is a neighborhood of $x$ on which $f \neq c$. If $f > c$ on this neighborhood, then we have $\max\{f, c\} = f$ on this neighborhood; if $f < c$, then we have $\max\{f, c\} = c$. In either case, this implies that $\max\{f, c\}$ is differentiable at $x$, and thus $x$ is not in $B$.

Next, consider the case where $f(x) = c$. First, suppose $f'(x) > 0$; then by Rolle's theorem, we can find a neighborhood of $x$ on which $f(x') > c$ if $x' > x$ and $f(x') < c$ if $x' < x$. Possibly shrinking this neighborhood, we can assume every point of the neighborhood is a point of differentiability of

$f$. Thus, for $x' < x$ in this neighborhood, we have $\max\{f(x'), c\} = f(x')$, and for $x' > x$, we have $\max\{f(x'), c\} = c$. We conclude that $\max\{f, c\}$ is differentiable at all points of this neighborhood except $x$, and in particular $x$ is an isolated point in $B$. A symmetric argument treats the case where $f'(x) < 0$, with the same conclusion.

On the other hand, if $f'(x) = 0$, we can write $f(x') = c + o(|x' - x|)$ for $x'$ in a neighborhood of $x$, which implies $\max\{f(x'), c\} = \max\{c, c + o(|x' - x|)\} = c \pm o(|x' - x|)$. In particular, $|\max\{f(x'), c\} - \max\{f(x), c\}| = o(|x' - x|)$, which shows that $\max\{f, c\}$ is differentiable at $x$, and thus $x$ is not in $B$. This shows that every point of $A \cap B$ is isolated in $A \cap B$, and we can therefore conclude that $\max\{f, c\}$ is differentiable except at isolated points of $(a, b)$. $\qquad\square$

**Lemma E.22.** *For $0 \le \nu \le \pi$, consider the function*

$$\tilde{\varphi}(\nu) = \mathop{\mathbb{E}}_{\boldsymbol{g}_1, \boldsymbol{g}_2 \sim \text{i.i.d.} \mathcal{N}(\boldsymbol{0}, (2/n)\boldsymbol{I})} [\mathbb{1}_{\mathcal{E}_1} \phi(\nu, \boldsymbol{g}_1, \boldsymbol{g}_2)],$$

*where*

$$\phi(\nu, \boldsymbol{g}_1, \boldsymbol{g}_2) = \cos^{-1}\left(\frac{\langle \boldsymbol{v}_0, \boldsymbol{v}_\nu \rangle}{\|\boldsymbol{v}_0\|_2 \|\boldsymbol{v}_\nu\|_2}\right).$$

*Then $\tilde{\varphi}$ is absolutely continuous on $[0, \pi]$, and satisfies the first-order Taylor expansion*

$$\tilde{\varphi}(\nu) = \tilde{\varphi}(0) - \int_0^\nu \mathop{\mathbb{E}}_{\boldsymbol{g}_1, \boldsymbol{g}_2} \left[ \mathbb{1}_{\mathcal{E}_1} \frac{\left\langle \frac{\boldsymbol{v}_0}{\|\boldsymbol{v}_0\|_2}, \frac{1}{\|\boldsymbol{v}_t\|_2} \left(\boldsymbol{I} - \frac{\boldsymbol{v}_t \boldsymbol{v}_t^*}{\|\boldsymbol{v}_t\|_2^2}\right) \dot{\boldsymbol{v}}_t \right\rangle}{\sqrt{1 - \left\langle \frac{\boldsymbol{v}_0}{\|\boldsymbol{v}_0\|_2}, \frac{\boldsymbol{v}_t}{\|\boldsymbol{v}_t\|_2} \right\rangle^2}} \right] dt,$$

*and moreover $\tilde{\varphi}$ is $1$-Lipschitz.*

*Proof.* At points of $(0, \pi)$ where each of the functions composed in $\phi$ is differentiable, the chain rule gives for the derivative of the integrand as a function of $\nu$

$$\phi'(\nu, \boldsymbol{g}_1, \boldsymbol{g}_2) = -\frac{\left\langle \frac{\boldsymbol{v}_0}{\|\boldsymbol{v}_0\|_2}, \frac{1}{\|\boldsymbol{v}_\nu\|_2} \left(\boldsymbol{I} - \frac{\boldsymbol{v}_\nu \boldsymbol{v}_\nu^*}{\|\boldsymbol{v}_\nu\|_2^2}\right) \dot{\boldsymbol{v}}_\nu \right\rangle}{\sqrt{1 - \left\langle \frac{\boldsymbol{v}_0}{\|\boldsymbol{v}_0\|_2}, \frac{\boldsymbol{v}_\nu}{\|\boldsymbol{v}_\nu\|_2} \right\rangle^2}}, \tag{E.38}$$

where we have used the result

$$\mathrm{d}\left(\frac{\cdot}{\|\cdot\|}\right)_{\boldsymbol{x}} = \frac{1}{\|\boldsymbol{x}\|_2}\left(\boldsymbol{I} - \frac{\boldsymbol{x}\boldsymbol{x}^*}{\|\boldsymbol{x}\|_2^2}\right),$$

valid for any $\boldsymbol{x} \ne \boldsymbol{0}$. Because $\mathcal{E}_1$ guarantees that $\boldsymbol{v}_\nu \ne \boldsymbol{0}$ for all $\nu \in [0, \pi]$, we see that the integrand $\phi$ is continuous. Similarly, given that $\|\boldsymbol{v}_\nu\|_2 \ge \frac{1}{2}$ on $\mathcal{E}_1$, we note that there are just two obstructions to differentiability:

1. The inverse cosine is not differentiable at $\{\pm 1\}$;

2. The activation $\sigma$ is not differentiable at $0$.

First we characterize the issue of nondifferentiability with regards to the inverse cosine. We note that $\cos \phi(\nu, \boldsymbol{g}_1, \boldsymbol{g}_2) = 1$ if and only if the Cauchy-Schwarz inequality is tight, which is equivalent to $\boldsymbol{v}_0$ and $\boldsymbol{v}_\nu$ being linearly dependent. Suppose we have $(\boldsymbol{g}_1, \boldsymbol{g}_2) \in \mathcal{E}_1$ and $\nu_0 \in (0, \pi)$ such that $\boldsymbol{v}_0(\boldsymbol{g}_1, \boldsymbol{g}_2)$ and $\boldsymbol{v}_{\nu_0}(\boldsymbol{g}_1, \boldsymbol{g}_2)$ are linearly dependent. Because two vectors $\boldsymbol{u}_1, \boldsymbol{u}_2 \in \mathbb{R}^n$ have $\sigma(\boldsymbol{u}_1)$ and $\sigma(\boldsymbol{u}_2)$ linearly dependent if and only if $\sigma(\boldsymbol{u}_1)$ and $\sigma(\boldsymbol{u}_2)$ have the same support and are linearly dependent on the support, and given that $\|\boldsymbol{v}_\nu\|_0 > 1$ for each $\nu$, we have that there is a $2 \times 2$ submatrix of $\boldsymbol{GM}_{\nu_0}$ having positive entries and rank $1$ (since the rank is zero if and only if the submatrix is zero), where

$$\boldsymbol{M}_\nu = \begin{bmatrix} 1 & \cos \nu \\ 0 & \sin \nu \end{bmatrix}.$$

Write the corresponding $2 \times 2$ submatrix of $\boldsymbol{G}$ as $\boldsymbol{X}$. Because $\operatorname{rank} \boldsymbol{M}_{\nu_0} = 2$ by $\nu_0 \in (0, \pi)$, we have $\operatorname{rank} \boldsymbol{X} = 1$. On the other hand, if $\boldsymbol{G} \sim_{\text{i.i.d.}} \mathcal{N}(0, 2/n)$, we have

$$
\mathbb{P}[\boldsymbol{G} \text{ has a singular } 2 \times 2 \text{ minor}] \leq \sum_{1 \leq i < j \leq n} \mathbb{P}\left[\operatorname{rank}\left(\begin{bmatrix} G_{1i} & G_{2i} \\ G_{1j} & G_{2j} \end{bmatrix}\right) < 2\right]
$$
$$
= 0,
$$

where the first line is a union bound, and the second line uses the fact that $2 \times 2$ submatrices of $\boldsymbol{G}$ are i.i.d. $\mathcal{N}(0, 2/n)$, and that the complement of the set of full-rank $2 \times 2$ matrices is a positive-codimensional closed embedded submanifold of $\mathbb{R}^{2 \times 2}$. It follows that the subset of $\mathcal{E}_1$ of matrices having no singular $2 \times 2$ minor has full measure in $\mathcal{E}_1$, and we conclude that for almost all $(\boldsymbol{g}_1, \boldsymbol{g}_2)$, we have $\cos \phi(\nu, \boldsymbol{g}_1, \boldsymbol{g}_2) < 1$ for every $\nu \in (0, \pi)$. Next, we characterize nondifferentiability due to the activation $\sigma$; by the chain rule, it suffices to consider nondifferentiability of $\boldsymbol{v}_\nu$ as a function of $\nu$, and then Lemma E.21 implies that for every $(\boldsymbol{g}_1, \boldsymbol{g}_2)$, $\boldsymbol{v}_\nu$ is differentiable at all but at most countably many points of $[0, \pi]$. Next, we observe that whenever $\boldsymbol{v}_\nu$ is nonvanishing, one has

$$
\left|\left\langle \frac{\boldsymbol{v}_0}{\|\boldsymbol{v}_0\|_2}, \frac{1}{\|\boldsymbol{v}_\nu\|_2}\left(\boldsymbol{I} - \frac{\boldsymbol{v}_\nu \boldsymbol{v}_\nu^*}{\|\boldsymbol{v}_\nu\|_2^2}\right)\dot{\boldsymbol{v}}_\nu \right\rangle\right| \leq \frac{\|\boldsymbol{P}_{\boldsymbol{v}_\nu}^\perp \dot{\boldsymbol{v}}_\nu\|_2}{\|\boldsymbol{v}_\nu\|_2}\left\|\left(\boldsymbol{I} - \frac{\boldsymbol{v}_\nu \boldsymbol{v}_\nu^*}{\|\boldsymbol{v}_\nu\|_2^2}\right)\frac{\boldsymbol{v}_0}{\|\boldsymbol{v}_0\|_2}\right\|_2
$$
$$
= \frac{\|\boldsymbol{P}_{\boldsymbol{v}_\nu}^\perp \dot{\boldsymbol{v}}_\nu\|_2}{\|\boldsymbol{v}_\nu\|_2}\sqrt{1 - \left\langle \frac{\boldsymbol{v}_0}{\|\boldsymbol{v}_0\|_2}, \frac{\boldsymbol{v}_\nu}{\|\boldsymbol{v}_\nu\|_2}\right\rangle^2},
$$

where the first inequality is due squaring the orthogonal projection and Cauchy-Schwarz, and the second equality follows from distributing to evaluate the squared norm, cancelling, and taking square roots. Using the fact that orthogonal projections have operator norm 1, we thus conclude

$$
|\phi'(\nu, \boldsymbol{g}_1, \boldsymbol{g}_2)| \leq \frac{\|\boldsymbol{P}_{\boldsymbol{v}_\nu}^\perp \dot{\boldsymbol{v}}_\nu\|_2}{\|\boldsymbol{v}_\nu\|_2} \leq C\|\dot{\boldsymbol{v}}_\nu\|_2, \tag{E.39}
$$

where the last inequality is valid whenever $(\boldsymbol{g}_1, \boldsymbol{g}_2) \in \mathcal{E}_1$. Since

$$
\|\dot{\boldsymbol{v}}_\nu\|_2 = \|\dot{\sigma}(\boldsymbol{g}_1 \cos \nu + \boldsymbol{g}_2 \sin \nu) \odot (\boldsymbol{g}_2 \cos \nu - \boldsymbol{g}_1 \sin \nu)\|_2
$$
$$
\leq \|\boldsymbol{g}_2 \cos \nu - \boldsymbol{g}_1 \sin \nu\|_2
$$
$$
\leq \|\boldsymbol{g}_2\| + \|\boldsymbol{g}_1\|_2,
$$

and this upper bound is jointly integrable in $\nu$ and $(\boldsymbol{g}_1, \boldsymbol{g}_2)$ over $[0, \pi] \times \mathbb{R}^n \times \mathbb{R}^n$, we can apply (Cohn, 2013, Theorem 6.3.11) to obtain that whenever $(\boldsymbol{g}_1, \boldsymbol{g}_2) \in \mathcal{E}_1$ minus a negligible set, we have for every $\nu \in [0, \pi]$

$$
\phi(\nu, \boldsymbol{g}_1, \boldsymbol{g}_2) = \phi(0, \boldsymbol{g}_1, \boldsymbol{g}_2) + \int_0^\nu \phi'(t, \boldsymbol{g}_1, \boldsymbol{g}_2)\, \mathrm{d}t.
$$

In particular, multiplying by the indicator for $\mathcal{E}_1$, taking expectations over $(\boldsymbol{g}_1, \boldsymbol{g}_2)$, and applying the previous joint integrability assertion for $\phi'$ together with Fubini's theorem yields

$$
\tilde{\varphi}(\nu) = \tilde{\varphi}(0) + \int_0^\nu \mathop{\mathbb{E}}_{\boldsymbol{g}_1, \boldsymbol{g}_2}[\phi'(t, \boldsymbol{g}_1, \boldsymbol{g}_2)]\, \mathrm{d}t,
$$

so to conclude the Lipschitz estimate, it suffices to obtain a suitable estimate on $\mathbb{E}_{\boldsymbol{g}_1,\boldsymbol{g}_2}[\phi'(\nu, \boldsymbol{g}_1, \boldsymbol{g}_2)]$. In light of (E.39) we calculate more precisely

$$
\begin{aligned}
\mathbb{E}\left[\mathbb{1}_{\mathcal{E}_1} \frac{\|\boldsymbol{P}_{\boldsymbol{v}_\nu}^\perp \dot{\boldsymbol{v}}_\nu\|_2}{\|\boldsymbol{v}_\nu\|_2}\right] &= \mathbb{E}\left[\mathbb{1}_{\mathcal{E}_1} \frac{\|\boldsymbol{P}_{\boldsymbol{v}_0}^\perp \dot{\boldsymbol{v}}_0\|_2}{\|\boldsymbol{v}_0\|_2}\right] \\
&= \mathbb{E}\left[\mathbb{1}_{\mathcal{E}_1} \frac{\left\|\left(\boldsymbol{I} - \frac{\sigma(\boldsymbol{g}_1)\sigma(\boldsymbol{g}_1)^*}{\|\sigma(\boldsymbol{g}_1)\|_2^2}\right)(\dot{\sigma}(\boldsymbol{g}_1) \odot \boldsymbol{g}_2)\right\|_2}{\|\sigma(\boldsymbol{g}_1)\|_2}\right] \\
&\leq \mathbb{E}\left[\mathbb{1}_{\|\sigma(\boldsymbol{g}_1)\|_0>1} \frac{\left\|\left(\boldsymbol{I} - \frac{\sigma(\boldsymbol{g}_1)\sigma(\boldsymbol{g}_1)^*}{\|\sigma(\boldsymbol{g}_1)\|_2^2}\right)(\dot{\sigma}(\boldsymbol{g}_1) \odot \boldsymbol{g}_2)\right\|_2}{\|\sigma(\boldsymbol{g}_1)\|_2}\right] \\
&= \sum_{k=2}^n 2^{-n}\binom{n}{k}\mathbb{E}\left[\frac{\left\|\left(\boldsymbol{I} - \frac{\sigma(\boldsymbol{g}_1)\sigma(\boldsymbol{g}_1)^*}{\|\sigma(\boldsymbol{g}_1)\|_2^2}\right)(\dot{\sigma}(\boldsymbol{g}_1) \odot \boldsymbol{g}_2)\right\|_2}{\|\sigma(\boldsymbol{g}_1)\|_2}\;\middle|\;\|\sigma(\boldsymbol{g}_1)\|_0 = k\right] \\
&= \sum_{k=2}^n 2^{-n}\binom{n}{k}\underset{X \sim \chi(k-1)}{\mathbb{E}}[X]\underset{Y \sim \chi(k)}{\mathbb{E}}\left[\frac{1}{Y}\right].
\end{aligned}
$$

In the first line, we apply rotational invariance and unpack notation; in the second line, we use non-negativity of the integrand to pass to the containing event where $\boldsymbol{v}_0$ is at least 2-sparse; and in the third line, we condition on the size of the support of $\boldsymbol{g}_1$. In the fourth line, we use several facts; first, we note that $\boldsymbol{P}_{\boldsymbol{v}_0}^\perp(\dot{\sigma}(\boldsymbol{g}_1) \odot \boldsymbol{g}_2) = \boldsymbol{P}_{\boldsymbol{v}_0}^\perp \boldsymbol{P}_{\{\sigma(\boldsymbol{g}_1)>0\}}\boldsymbol{g}_2$ for any $\boldsymbol{g}_2 \in \mathbb{R}^n$, and that the commutation relation $\boldsymbol{P}_{\boldsymbol{v}_0}^\perp \boldsymbol{P}_{\{\sigma(\boldsymbol{g}_1)>0\}} = \boldsymbol{P}_{\{\sigma(\boldsymbol{g}_1)>0\}}\boldsymbol{P}_{\boldsymbol{v}_0}^\perp$ implies that the operator $\boldsymbol{P}_{\boldsymbol{v}_0}^\perp \boldsymbol{P}_{\{\sigma(\boldsymbol{g}_1)>0\}}$ is itself an orthogonal projection, with range equal to the $(\|\boldsymbol{v}_0\|_0 - 1)$-dimensional subspace consisting of vectors with support $\mathrm{supp}(\boldsymbol{v}_0)$ orthogonal to $\boldsymbol{v}_0$. In particular, $\sigma(\boldsymbol{g}_1)$ and $\boldsymbol{P}_{\boldsymbol{v}_0}^\perp \boldsymbol{P}_{\{\boldsymbol{v}_0>0\}}\boldsymbol{g}_2$ are independent gaussian vectors, and conditioned on the size of the support of $\sigma(\boldsymbol{g}_1)$ the quantities $\|\sigma(\boldsymbol{g}_1)\|_2$ and $\|\boldsymbol{P}_{\boldsymbol{v}_0}^\perp \boldsymbol{P}_{\{\boldsymbol{v}_0>0\}}\boldsymbol{g}_2\|_2$ are distributed as independent chi random variables with (respectively) $k$ and $k - 1$ degrees of freedom. An application of Lemma G.9 then gives

$$
\mathbb{E}\left[\mathbb{1}_{\mathcal{E}_1} \frac{\|\boldsymbol{P}_{\boldsymbol{v}_\nu}^\perp \dot{\boldsymbol{v}}_\nu\|_2}{\|\boldsymbol{v}_\nu\|_2}\right] \leq 1, \tag{E.40}
$$

which is sufficient to conclude. $\qquad\square$

**Lemma E.23.** *The random variable $X_\nu$ satisfies the following regularity properties:*

1. *If $0 < \nu \leq \pi$, we have $X_\nu < 1$ almost surely.*

2. *If $(\boldsymbol{g}_1, \boldsymbol{g}_2) \in \mathcal{E}_1$, then $X_\nu$ is absolutely continuous on $[0, \pi]$, with a.e. derivative*

$$
\dot{X}_\nu = \left\langle \frac{\boldsymbol{v}_0}{\|\boldsymbol{v}_0\|_2}, \frac{1}{\|\boldsymbol{v}_\nu\|_2}\left(\boldsymbol{I} - \frac{\boldsymbol{v}_\nu \boldsymbol{v}_\nu^*}{\|\boldsymbol{v}_\nu\|_2^2}\right)\dot{\boldsymbol{v}}_\nu \right\rangle,
$$

*and moreover we have $\mathbb{E}_{\boldsymbol{g}_1,\boldsymbol{g}_2}[|\dot{X}_\nu|] \leq 1$, so the analogous differentiation result applies to $\mathbb{E}_{\boldsymbol{g}_1,\boldsymbol{g}_2}[X_\nu]$.*

*Proof.* The first claim is a corollary of the proof of differentiability of the inverse cosine part of $\tilde{\varphi}$ in Lemma E.22 and the observation that $X_\pi = 0$. The second claim is also a direct consequence of the proof of Lemma E.22 and Fubini's theorem. $\qquad\square$

**Lemma E.24.** *Consider the function*

$$
f(\nu) = \underset{\boldsymbol{g}_1,\boldsymbol{g}_2}{\mathbb{E}}[X_\nu] = \underset{\boldsymbol{g}_1,\boldsymbol{g}_2}{\mathbb{E}}\left[\mathbb{1}_{\mathcal{E}_1}\left\langle \frac{\boldsymbol{v}_0}{\|\boldsymbol{v}_0\|_2}, \frac{\boldsymbol{v}_\nu}{\|\boldsymbol{v}_\nu\|_2}\right\rangle\right].
$$

*Then $f$ is continuously differentiable, with derivative*

$$
f'(\nu) = \underset{\boldsymbol{g}_1,\boldsymbol{g}_2}{\mathbb{E}}\left[\mathbb{1}_{\mathcal{E}_1}\left\langle \frac{\boldsymbol{v}_0}{\|\boldsymbol{v}_0\|_2}, \frac{1}{\|\boldsymbol{v}_\nu\|_2}\left(\boldsymbol{I} - \frac{\boldsymbol{v}_\nu \boldsymbol{v}_\nu^*}{\|\boldsymbol{v}_\nu\|_2^2}\right)\dot{\boldsymbol{v}}_\nu\right\rangle\right].
$$

*Moreover, $f'$ is absolutely continuous, with Lebesgue-a.e. derivative*

$$f''(\nu) = -\mathop{\mathbb{E}}_{\boldsymbol{g}_1,\boldsymbol{g}_2}\left[\mathbb{1}_{\mathcal{E}_1}\left\langle \frac{1}{\|\boldsymbol{v}_\nu\|_2}\left(\boldsymbol{I} - \frac{\boldsymbol{v}_\nu \boldsymbol{v}_\nu^*}{\|\boldsymbol{v}_\nu\|_2^2}\right)\dot{\boldsymbol{v}}_\nu, \frac{1}{\|\boldsymbol{v}_0\|_2}\left(\boldsymbol{I} - \frac{\boldsymbol{v}_0 \boldsymbol{v}_0^*}{\|\boldsymbol{v}_0\|_2^2}\right)\dot{\boldsymbol{v}}_0\right\rangle\right]$$

*Proof.* The expression for $f'$ is a direct consequence of Lemma E.23. To see that $f'$ is actually continuous, apply rotational invariance of the Gaussian measure and of $\mathbb{1}_{\mathcal{E}_1}$ by Lemma E.16 to get

$$f'(\nu) = -\mathop{\mathbb{E}}_{\boldsymbol{g}_1,\boldsymbol{g}_2}\left[\mathbb{1}_{\mathcal{E}_1}\left\langle \frac{\boldsymbol{v}_\nu}{\|\boldsymbol{v}_\nu\|_2}, \frac{1}{\|\boldsymbol{v}_0\|_2}\left(\boldsymbol{I} - \frac{\boldsymbol{v}_0 \boldsymbol{v}_0^*}{\|\boldsymbol{v}_0\|_2^2}\right)\dot{\boldsymbol{v}}_0\right\rangle\right],$$

then notice that this expression is an integral of a continuous function of $\nu$, which is therefore continuous. Moreover, the $\nu$ dependence in this expression for $f'$ mirrors exactly that of $f$; in particular, the integrand

$$-\left\langle \frac{\boldsymbol{v}_\nu}{\|\boldsymbol{v}_\nu\|_2}, \frac{1}{\|\boldsymbol{v}_0\|_2}\left(\boldsymbol{I} - \frac{\boldsymbol{v}_0 \boldsymbol{v}_0^*}{\|\boldsymbol{v}_0\|_2^2}\right)\dot{\boldsymbol{v}}_0\right\rangle$$

is absolutely continuous whenever $(\boldsymbol{g}_1,\boldsymbol{g}_2)\in\mathcal{E}_1$ by Lemma E.23, with a.e. derivative

$$-\left\langle \frac{1}{\|\boldsymbol{v}_\nu\|_2}\left(\boldsymbol{I} - \frac{\boldsymbol{v}_\nu \boldsymbol{v}_\nu^*}{\|\boldsymbol{v}_\nu\|_2^2}\right)\dot{\boldsymbol{v}}_\nu, \frac{1}{\|\boldsymbol{v}_0\|_2}\left(\boldsymbol{I} - \frac{\boldsymbol{v}_0 \boldsymbol{v}_0^*}{\|\boldsymbol{v}_0\|_2^2}\right)\dot{\boldsymbol{v}}_0\right\rangle.$$

We can therefore conclude the claimed expression for $f''$ provided we can show absolute integrability over $\mathcal{E}_1$ of this last expression, using Fubini's theorem in a way analogous to the argument in Lemma E.22. But

$$\mathop{\mathbb{E}}_{\boldsymbol{g}_1,\boldsymbol{g}_2}\left[\mathbb{1}_{\mathcal{E}_1}\left|\left\langle \frac{\dot{\boldsymbol{v}}_\nu}{\|\boldsymbol{v}_\nu\|_2}\left(\boldsymbol{I} - \frac{\boldsymbol{v}_\nu \boldsymbol{v}_\nu^*}{\|\boldsymbol{v}_\nu\|_2^2}\right), \frac{\dot{\boldsymbol{v}}_0}{\|\boldsymbol{v}_0\|_2}\left(\boldsymbol{I} - \frac{\boldsymbol{v}_0 \boldsymbol{v}_0^*}{\|\boldsymbol{v}_0\|_2^2}\right)\right\rangle\right|\right] \le 4\mathop{\mathbb{E}}_{\boldsymbol{g}_1,\boldsymbol{g}_2}\left[\mathbb{1}_{\mathcal{E}_1}\left\|\boldsymbol{P}_{\boldsymbol{v}_\nu}^\perp \dot{\boldsymbol{v}}_\nu\right\|_2\left\|\boldsymbol{P}_{\boldsymbol{v}_0}^\perp \dot{\boldsymbol{v}}_0\right\|_2\right]$$

$$\le 4\mathbb{E}\left[\left\|\dot{\boldsymbol{v}}_0\right\|_2^2\right] = 4,$$

using, in sequence, Cauchy-Schwarz and the lower bound in the definition of $\mathcal{E}_1$; the operator norm of orthogonal projections being 1, the Schwarz inequality, nonnegativity of the integrand, and rotational invariance; and Lemma E.17. We can therefore conclude the claimed expression for $f''$ and complete the proof. □

**Lemma E.25.** *For the heuristic cosine angle evolution function*

$$\cos\varphi(\nu) = \mathop{\mathbb{E}}_{\boldsymbol{g}_1,\boldsymbol{g}_2}\left[\langle\boldsymbol{v}_0,\boldsymbol{v}_\nu\rangle\right],$$

*we have the following integral representations for its continuous derivatives:*

$$(\cos\circ\varphi)'(\nu) = \mathop{\mathbb{E}}_{\boldsymbol{g}_1,\boldsymbol{g}_2}\left[\langle\boldsymbol{v}_0,\dot{\boldsymbol{v}}_\nu\rangle\right]$$

$$(\cos\circ\varphi)''(\nu) = -\mathop{\mathbb{E}}_{\boldsymbol{g}_1,\boldsymbol{g}_2}\left[\langle\dot{\boldsymbol{v}}_0,\dot{\boldsymbol{v}}_\nu\rangle\right].$$

*Proof.* The proof follows exactly the arguments of Lemma E.24, but with a simpler integrand and different integrability checks; the continuity assertion relies on Lemma E.5. Indeed, this approach gives that $\langle\boldsymbol{v}_0,\boldsymbol{v}_\nu\rangle$ is absolutely continuous, with Lebesgue-a.e. derivative $\langle\boldsymbol{v}_0,\dot{\boldsymbol{v}}_\nu\rangle$; we check

$$\mathop{\mathbb{E}}_{\boldsymbol{g}_1,\boldsymbol{g}_2}\left[|\langle\boldsymbol{v}_0,\dot{\boldsymbol{v}}_\nu\rangle|\right] \le \mathop{\mathbb{E}}_{\boldsymbol{g}_1,\boldsymbol{g}_2}\left[\|\boldsymbol{v}_0\|_2^2\right]^{1/2}\mathop{\mathbb{E}}_{\boldsymbol{g}_1,\boldsymbol{g}_2}\left[\|\dot{\boldsymbol{v}}_0\|_2^2\right]^{1/2} \le 1$$

by Cauchy-Schwarz, the Schwarz inequality, rotational invariance, and Lemma E.17. This verifies the claimed expression for $(\cos\circ\varphi)'$. For the second derivative, we apply rotational invariance to get

$$(\cos\circ\varphi)'(\nu) = -\mathop{\mathbb{E}}_{\boldsymbol{g}_1,\boldsymbol{g}_2}\left[\langle\boldsymbol{v}_\nu,\dot{\boldsymbol{v}}_0\rangle\right],$$

which has an absolutely continuous integrand, with Lebesgue-a.e. derivative

$$-\langle\dot{\boldsymbol{v}}_0,\dot{\boldsymbol{v}}_\nu\rangle.$$

Checking absolute integrability, we have as before

$$\mathop{\mathbb{E}}_{\boldsymbol{g}_1,\boldsymbol{g}_2}\left[|\langle\dot{\boldsymbol{v}}_0,\dot{\boldsymbol{v}}_\nu\rangle|\right] \le \mathop{\mathbb{E}}_{\boldsymbol{g}_1,\boldsymbol{g}_2}\left[\|\dot{\boldsymbol{v}}_0\|_2^2\right] \le 1$$

by Cauchy-Schwarz, the Schwarz inequality, rotational invariance, and Lemma E.17. This establishes the claimed expression for $(\cos\circ\varphi)''$. □

**Lemma E.26.** *Let* $\psi : \mathbb{R} \to \mathbb{R}$ *be defined by* $\psi(x) = \psi_{0.25}(x)$*, where* $\psi_{0.25}$ *is the function constructed in Lemma E.31. Then the function*

$$f(\nu, \boldsymbol{g}_1) = \mathop{\mathbb{E}}_{\boldsymbol{g}_2}\left[\frac{\langle \boldsymbol{v}_0, \boldsymbol{v}_\nu \rangle}{\psi(\|\boldsymbol{v}_0\|_2)\psi(\|\boldsymbol{v}_\nu\|_2)}\right]$$

*satisfies for all* $\nu \in [0, \pi]$ *and Lebesgue-a.e.* $\boldsymbol{g}_1$ *the second-order Taylor expansion*

$$f(\nu, \boldsymbol{g}_1) = \frac{\|\boldsymbol{v}_0\|_2^2}{\psi(\|\boldsymbol{v}_0\|_2)^2} + \int_0^\nu \int_0^t \left( \mathop{\mathbb{E}}_{\boldsymbol{g}_2}\left[ \sum_{i=1}^n \frac{\sigma(g_{1i})^3 \rho(-g_{1i}\cot s)}{\psi(\|\boldsymbol{v}_0\|_2)\psi(\|\boldsymbol{v}_s^i\|_2)\sin^3 s}\right] \right.$$
$$- \mathop{\mathbb{E}}_{\boldsymbol{g}_2}\left[ \frac{\langle \boldsymbol{v}_0, \boldsymbol{v}_s \rangle}{\psi(\|\boldsymbol{v}_0\|_2)\psi(\|\boldsymbol{v}_s\|_2)} - \frac{\langle \boldsymbol{v}_0, \boldsymbol{v}_s \rangle \psi'(\|\boldsymbol{v}_s\|_2)\|\boldsymbol{v}_s\|_2}{\psi(\|\boldsymbol{v}_0\|_2)\psi(\|\boldsymbol{v}_s\|_2)^2}\right]$$
$$- \mathop{\mathbb{E}}_{\boldsymbol{g}_2}\left[ + \frac{\langle \boldsymbol{v}_0, \boldsymbol{v}_s \rangle \langle \boldsymbol{v}_s, \dot{\boldsymbol{v}}_s \rangle^2 \psi''(\|\boldsymbol{v}_s\|_2)}{\psi(\|\boldsymbol{v}_0\|_2)\psi(\|\boldsymbol{v}_s\|_2)^2 \|\boldsymbol{v}_s\|_2^2}\right]$$
$$+ \mathop{\mathbb{E}}_{\boldsymbol{g}_2}\left[ -2\frac{\langle \boldsymbol{v}_0, \dot{\boldsymbol{v}}_s \rangle \langle \boldsymbol{v}_s, \dot{\boldsymbol{v}}_s \rangle \psi'(\|\boldsymbol{v}_s\|_2)}{\psi(\|\boldsymbol{v}_0\|_2)\psi(\|\boldsymbol{v}_s\|_2)^2 \|\boldsymbol{v}_s\|_2} - \frac{\langle \boldsymbol{v}_0, \boldsymbol{v}_s \rangle \|\dot{\boldsymbol{v}}_s\|_2^2 \psi'(\|\boldsymbol{v}_s\|_2)}{\psi(\|\boldsymbol{v}_0\|_2)\psi(\|\boldsymbol{v}_s\|_2)^2 \|\boldsymbol{v}_s\|_2}\right]$$
$$\left. + \mathop{\mathbb{E}}_{\boldsymbol{g}_2}\left[ 2\frac{\langle \boldsymbol{v}_0, \boldsymbol{v}_s \rangle \langle \boldsymbol{v}_s, \dot{\boldsymbol{v}}_s \rangle^2 \psi'(\|\boldsymbol{v}_s\|_2)}{\psi(\|\boldsymbol{v}_0\|_2)\psi(\|\boldsymbol{v}_s\|_2)^3 \|\boldsymbol{v}_s\|_2^2} + \frac{\langle \boldsymbol{v}_0, \boldsymbol{v}_s \rangle \langle \boldsymbol{v}_s, \dot{\boldsymbol{v}}_s \rangle^2 \psi'(\|\boldsymbol{v}_s\|_2)}{\psi(\|\boldsymbol{v}_0\|_2)\psi(\|\boldsymbol{v}_s\|_2)^2 \|\boldsymbol{v}_s\|_2^3}\right] \right) \, \mathrm{d}s.$$

*where previously-unspecified notation in this expression is introduced in* (E.44)*.*

*Proof.* Take $\boldsymbol{g}_1 \in \mathbb{R}^n$ such that $f(\nu, \cdot)$ exists and is $\boldsymbol{g}_1$-integrable; by Fubini's theorem such $\boldsymbol{g}_1$ have full measure in $\mathbb{R}^n$. Because $\psi > 0$ and $\psi(\|\boldsymbol{v}_\nu\|)$ is locally (as a function of $\nu$) constant whenever $\|\boldsymbol{v}_\nu\| < \frac{1}{4}$, we need only consider nondifferentiability of $\sigma$ when assessing differentiability of $f(\cdot, \boldsymbol{g}_1)$. By Lemma E.21, we conclude that $f(\cdot, \boldsymbol{g}_1)$ is differentiable at all but at most countably many points of $(0, \pi)$; since $\psi > 0$ and $\psi$ is smooth, $f$ is continuous, and we can therefore apply Lebesgue differentiation theorems (Cohn, 2013, Theorem 6.3.11) to $f$ provided we satisfy the standard derivative product integrability checks. Writing

$$\phi(\nu, \boldsymbol{g}_1, \boldsymbol{g}_2) = \frac{\langle \boldsymbol{v}_0, \boldsymbol{v}_\nu \rangle}{\psi(\|\boldsymbol{v}_0\|_2)\psi(\|\boldsymbol{v}_\nu\|_2)},$$

the chain rule gives (at points of differentiability)

$$\phi'(\nu, \boldsymbol{g}_1, \boldsymbol{g}_2) = \left\langle \frac{\boldsymbol{v}_0}{\psi(\|\boldsymbol{v}_0\|_2)\psi(\|\boldsymbol{v}_\nu\|_2)}, \dot{\boldsymbol{v}}_\nu \right\rangle - \left\langle \frac{\langle \boldsymbol{v}_0, \boldsymbol{v}_\nu \rangle \psi'(\|\boldsymbol{v}_\nu\|_2)\boldsymbol{v}_\nu}{\psi(\|\boldsymbol{v}_0\|_2)\psi(\|\boldsymbol{v}_\nu\|_2)^2 \|\boldsymbol{v}_\nu\|_2}, \dot{\boldsymbol{v}}_\nu \right\rangle.$$

In this expression, we follow the convention $0/0 = 0$ to account for the possibility that $\|\boldsymbol{v}_\nu\|_2 = 0$ (in this case, the $\psi'$ term handles the denominator). For product integrability, we Lemma E.31 to get $|\psi'| \le C$ for some absolute constant $C > 0$ together with Cauchy-Schwarz and the triangle inequality to get

$$|\phi'(\nu, \boldsymbol{g}_1, \boldsymbol{g}_2)| \le 16\|\boldsymbol{v}_0\|_2 \|\dot{\boldsymbol{v}}_\nu\|_2 + 64C\|\boldsymbol{v}_0\|_2 \|\boldsymbol{v}_\nu\|_2 \|\dot{\boldsymbol{v}}_\nu\|_2,$$

and applying the Schwarz inequality, rotational invariance (to eliminate $\nu$ dependence in the resulting expectations) and Lemma E.17, we conclude that $\phi'$ is jointly absolutely integrable over $[0, \pi] \times (\mathbb{R}^{n\times 2}, \mu \otimes \mu)$. We have therefore a first-order Taylor expansion

$$f(\nu, \boldsymbol{g}_1) = f(0, \boldsymbol{g}_1)$$
$$+ \int_0^\nu \left( \underbrace{\mathop{\mathbb{E}}_{\boldsymbol{g}_2}\left[ \left\langle \frac{\boldsymbol{v}_0}{\psi(\|\boldsymbol{v}_0\|_2)\psi(\|\boldsymbol{v}_t\|_2)}, \dot{\boldsymbol{v}}_t \right\rangle \right]}_{\Xi_1(\nu)} - \underbrace{\mathop{\mathbb{E}}_{\boldsymbol{g}_2}\left[ \left\langle \frac{\langle \boldsymbol{v}_0, \boldsymbol{v}_t \rangle \psi'(\|\boldsymbol{v}_t\|_2)\boldsymbol{v}_t}{\psi(\|\boldsymbol{v}_0\|_2)\psi(\|\boldsymbol{v}_t\|_2)^2 \|\boldsymbol{v}_t\|_2}, \dot{\boldsymbol{v}}_t \right\rangle \right]}_{\Xi_2(\nu)} \right) \, \mathrm{d}t.$$

We have

$$f(0, \boldsymbol{g}_1) = \mathop{\mathbb{E}}_{\boldsymbol{g}_2}\left[ \frac{\|\boldsymbol{v}_0\|_2^2}{\psi(\|\boldsymbol{v}_0\|_2)^2} \right] = \frac{\|\boldsymbol{v}_0\|_2^2}{\psi(\|\boldsymbol{v}_0\|_2)^2},$$

since $\boldsymbol{v}_0$ depends only on $\boldsymbol{g}_1$. Next, we show $t$-differentiability of the inner expectation. Our aim is to apply Lemma E.27 to differentiate $\Xi_1$ and $\Xi_2$. We first focus on $\Xi_1$; distributing and applying linearity, we have

$$\Xi_1(\nu) = \sum_{i=1}^n \mathbb{E}_{\boldsymbol{g}_2} \left[ \frac{\sigma(g_{1i})(g_{2i}\cos\nu - g_{1i}\sin\nu)}{\psi(\|\boldsymbol{v}_0\|_2)\psi(\|\boldsymbol{v}_\nu\|_2)} \dot\sigma(g_{1i}\cos\nu + g_{2i}\sin\nu) \right].$$

We have shown absolute integrability of the quantity inside the expectation above; we can therefore apply Fubini's theorem and the previous definition to write

$$\Xi_1(\nu) = \sum_{i=1}^n \mathbb{E}_{(g_{2j}):j\neq i} \left[ \mathbb{E}_{g_{2i}} \left[ \frac{\sigma(g_{1i})(g_{2i}\cos\nu - g_{1i}\sin\nu)}{\psi(\|\boldsymbol{v}_0(\boldsymbol{g}_1,\boldsymbol{g}_2)\|_2)\psi(\|\boldsymbol{v}_\nu(\boldsymbol{g}_1,\boldsymbol{g}_2)\|_2)} \dot\sigma(g_{1i}\cos\nu + g_{2i}\sin\nu) \right] \right]. \quad \text{(E.41)}$$

For each $i \in [n]$, write $\pi_i : \mathbb{R}^n \to \mathbb{R}^{n-1}$ for the linear map that deletes the $i$-th coordinate from its input, and let $\hat\pi_i : \mathbb{R} \times \mathbb{R}^{n-1} \to \mathbb{R}^n$ be the linear map such that $\hat\pi_i(g_i, \pi_i(\boldsymbol{g})) = \boldsymbol{g}$. With $\boldsymbol{g}_2$ fixed (in the context of (E.41)), if we define

$$f_1(\nu, g) = \frac{\sigma(g_{1i})(g\cos\nu - g_{1i}\sin\nu)}{\psi(\|\boldsymbol{v}_0(\boldsymbol{g}_1, \hat\pi_i(g, \pi_i(\boldsymbol{g}_2)))\|_2)\psi(\|\boldsymbol{v}_\nu(\boldsymbol{g}_1, \hat\pi_i(g, \pi_i(\boldsymbol{g}_2)))\|_2)},$$

then we can write

$$\Xi_1(\nu) = \sum_{i=1}^n \mathbb{E}_{(g_{2j}):j\neq i} \left[ \mathbb{E}_{g_{2i}} \left[ f_1(\nu, g_{2i}) \dot\sigma(g_{1i}\cos\nu + g_{2i}\sin\nu) \right] \right].$$

Thus, to differentiate $\Xi_1$, it suffices to check the regularity of $f_1(\nu, g)$ and apply Lemma E.27. As before, $\psi > 0$ and $\psi$ smooth implies that $f_1$ is continuous on $[0, \pi] \times \mathbb{R}$. For integrability of $f$, we appeal to the Fubini's theorem justification that we applied previously. For absolute continuity, we apply Lemma E.21 to get that the derivative of $f$ with respect to $\nu$ is, by the chain rule,

$$f_1'(\nu, g) = -\sigma(g_{1i}) \left( \frac{g_{1i}\cos\nu + g\sin\nu}{\psi(\|\boldsymbol{v}_0\|_2)\psi(\|\boldsymbol{v}_\nu\|_2)} + \frac{(g\cos\nu - g_{1i}\sin\nu)\psi'(\|\boldsymbol{v}_\nu\|_2)\langle\boldsymbol{v}_\nu, \dot{\boldsymbol{v}}_\nu\rangle}{\psi(\|\boldsymbol{v}_0\|_2)\psi(\|\boldsymbol{v}_\nu\|_2)^2\|\boldsymbol{v}_\nu\|_2} \right)$$

at all but at most countably many values of $\nu$; and the triangle inequality, Cauchy-Schwarz, and Lemma E.31 yield

$$\begin{aligned}
|f_1'(\nu, g)| &\leq \sigma(g_{1i}) \left( 16(|g_{1i}| + |g|) + 64C(|g| + |g_{1i}|)\|\dot{\boldsymbol{v}}_\nu\|_2 \right) \\
&\leq \sigma(g_{1i})(|g| + |g_{1i}|) \left( 16 + 64C(\|\boldsymbol{g}_1\|_2 + \|\boldsymbol{g}_2\|_2) \right) \\
&\leq \sigma(g_{1i})(|g| + |g_{1i}|) \left( 16 + 64C(\|\boldsymbol{g}_1\|_2 + \|\pi_i(\boldsymbol{g}_2)\|_2 + |g|) \right), \quad \text{(E.42)}
\end{aligned}$$

(we apply square root subadditivity in the last line) which is jointly integrable over $[0, \pi] \times \mathbb{R}$, and moreover over $[0, \pi] \times \mathbb{R}^n$. We conclude absolute continuity of $f_1(\cdot, g)$ and the integrability property of $f_1'$. Finally, for the growth estimate, we obtain an estimate for $f_1$ similar to the one we just obtained for $f_1'$ as follows:

$$|f_1(\nu, g)| \leq 16|g_{1i}|(|g| + |g_{1i}|); \quad \text{(E.43)}$$

the RHS of the final inequality above is a linear function of $|g|$, and when $|g| \geq 1$ we can therefore obtain $|f_1(\nu, g)| \leq 16(|g_{1i}| + |g_{1i}|^2)|g|$, which is a suitable growth estimate with $p = 1$. Then as long as $g_{1i} \neq 0$ for all $i$ (such $\boldsymbol{g}_1$ form a set of measure zero, which we can neglect), we can apply Lemma E.27 to get

$$\Xi_1(\nu) = \sum_{i=1}^n \mathbb{E}_{(g_{2j}):j\neq i} \left[ \mathbb{E}_{g_{2i}} \left[ f_1(0, g_{2i})\dot\sigma(g_{1i}) \right] + \int_0^\nu \left( \begin{aligned} &\mathbb{E}_{g_{2i}}[f_1'(t, g_{2i})\dot\sigma(g_{1i}\cos t + g_{2i}\sin t)] \\ &-g_{1i}\frac{f_1(t, -g_{1i}\cot t)\rho(-g_{1i}\cot t)}{\sin^2 t} \end{aligned} \right) \mathrm{d}t \right].$$

The estimates (E.42) and (E.43) show, respectively, that $f_1'$ and $f_1$ are absolutely integrable functions of $(\nu, \boldsymbol{g}_2)$. We have

$$f_1\left(t, -g_{1i}\cot t\right) = -\frac{\sigma(g_{1i})^2}{\psi(\|\boldsymbol{v}_0(\boldsymbol{g}_1, \boldsymbol{g}_2)\|_2)\psi(\|\boldsymbol{v}_t(\boldsymbol{g}_1, \hat\pi_i(-g_{1i}\cot t, \pi_i(\boldsymbol{g}_2)))\|_2)\sin t},$$

so that Lemma E.31 and nonnegativity give

$$\left| g_{1i}\frac{f_1\left(t, -g_{1i}\cot t\right)\rho(-g_{1i}\cot t)}{\sin^2 t} \right| \leq 16\frac{\sigma(g_{1i})^3}{\sin^3 t}\rho(-g_{1i}\cot t).$$

As in the proof of Lemma E.37, in particular using the estimates (E.52) (E.53) to control the magnitude of the RHS for all values of $t$, we can conclude that the Dirac term is absolutely integrable over $[0, \pi] \times \mathbb{R}^n$. An application of Fubini's theorem then allows us to re-combine the split integrals in the previous expression:

$$\Xi_1(\nu) = \sum_{i=1}^{n} \mathbb{E}_{\boldsymbol{g}_2}[f_1(0, g_{2i})\dot{\sigma}(g_{1i})] + \int_0^\nu \left( \begin{array}{c} \mathbb{E}_{\boldsymbol{g}_2}[f_1'(t, g_{2i})\dot{\sigma}(g_{1i}\cos t + g_{2i}\sin t)] \\ -g_{1i}\frac{\rho(-g_{1i}\cot t)}{\sin^2 t}\mathbb{E}_{\boldsymbol{g}_2}[f_1(t, -g_{1i}\cot t)] \end{array} \right) \mathrm{d}t.$$

We notice that

$$\boldsymbol{v}_t(\boldsymbol{g}_1, \hat{\pi}_i(-g_{1i}\cot t, \pi_i(\boldsymbol{g}_2))) = \begin{bmatrix} \sigma(g_{11}\cos\nu + g_{21}\sin\nu) \\ \vdots \\ \sigma(g_{1(i-1)}\cos\nu + g_{2(i-1)}\sin\nu) \\ 0 \\ \sigma(g_{1(i+1)}\cos\nu + g_{2(i+1)}\sin\nu) \\ \vdots \\ \sigma(g_{1n}\cos\nu + g_{2n}\sin\nu) \end{bmatrix},$$

and thus motivated introduce the notation

$$\begin{aligned} \tilde{\boldsymbol{g}}^i(t, \boldsymbol{g}_1, \boldsymbol{g}_2) &= \hat{\pi}_i(-g_{1i}\cot t, \pi_i(\boldsymbol{g}_2)); \\ \boldsymbol{v}_t^i(\boldsymbol{g}_1, \boldsymbol{g}_2) &= \boldsymbol{v}_t(\boldsymbol{g}_1, \tilde{\boldsymbol{g}}^i(t, \boldsymbol{g}_1, \boldsymbol{g}_2)). \end{aligned} \tag{E.44}$$

We can then write

$$-g_{1i}\frac{f_1(t, -g_{1i}\cot t)\rho(-g_{1i}\cot t)}{\sin^2 t} = \frac{\sigma(g_{1i})^3\rho(-g_{1i}\cot t)}{\psi(\|\boldsymbol{v}_0\|_2)\psi(\|\boldsymbol{v}_t^i\|_2)\sin^3 t}.$$

Finally, we apply linearity of the integral to move the summation over $i$ back inside the integrals, obtaining

$$\begin{aligned} \Xi_1(\nu) = {}& \mathbb{E}_{\boldsymbol{g}_2}\left[ \frac{\langle \boldsymbol{v}_0, \dot{\boldsymbol{v}}_0 \rangle}{\psi(\|\boldsymbol{v}_0\|_2)^2} \right] \\ &+ \int_0^\nu \left( \mathbb{E}_{\boldsymbol{g}_2}\left[ \sum_{i=1}^n \frac{\sigma(g_{1i})^3\rho(-g_{1i}\cot t)}{\psi(\|\boldsymbol{v}_0\|_2)\psi(\|\boldsymbol{v}_t^i\|_2)\sin^3 t} \right] - \mathbb{E}_{\boldsymbol{g}_2}\left[ \left( \begin{array}{c} \frac{\langle \boldsymbol{v}_0, \boldsymbol{v}_t \rangle}{\psi(\|\boldsymbol{v}_0\|_2)\psi(\|\boldsymbol{v}_t\|_2)} \\ +\frac{\langle \boldsymbol{v}_0, \dot{\boldsymbol{v}}_t \rangle\langle \boldsymbol{v}_t, \dot{\boldsymbol{v}}_t \rangle\psi'(\|\boldsymbol{v}_t\|_2)}{\psi(\|\boldsymbol{v}_0\|_2)\psi(\|\boldsymbol{v}_t\|_2)^2\|\boldsymbol{v}_t\|_2} \end{array} \right) \right] \right) \mathrm{d}t. \end{aligned}$$

Noting that, in the zero-order term, the only $\boldsymbol{g}_2$ dependence is in $\dot{\boldsymbol{v}}_0 = \dot{\sigma}(\boldsymbol{g}_1) \odot \boldsymbol{g}_2$, we apply independence of $\boldsymbol{g}_1$ and $\boldsymbol{g}_2$ to obtain finally

$$\begin{aligned} \Xi_1(\nu) = {}& \int_0^\nu \mathbb{E}_{\boldsymbol{g}_2}\left[ \sum_{i=1}^n \frac{\sigma(g_{1i})^3\rho(-g_{1i}\cot t)}{\psi(\|\boldsymbol{v}_0\|_2)\psi(\|\boldsymbol{v}_t^i\|_2)\sin^3 t} \right] \mathrm{d}t \\ &- \int_0^\nu \mathbb{E}_{\boldsymbol{g}_2}\left[ \left( \frac{\langle \boldsymbol{v}_0, \boldsymbol{v}_t \rangle}{\psi(\|\boldsymbol{v}_0\|_2)\psi(\|\boldsymbol{v}_t\|_2)} + \frac{\langle \boldsymbol{v}_0, \dot{\boldsymbol{v}}_t \rangle\langle \boldsymbol{v}_t, \dot{\boldsymbol{v}}_t \rangle\psi'(\|\boldsymbol{v}_t\|_2)}{\psi(\|\boldsymbol{v}_0\|_2)\psi(\|\boldsymbol{v}_t\|_2)^2\|\boldsymbol{v}_t\|_2} \right) \right] \mathrm{d}t. \end{aligned}$$

We run the same type of argument on $\Xi_2$ next. Distributing and applying linearity, we have

$$\Xi_2(\nu) = \sum_{i=1}^n \mathbb{E}_{\boldsymbol{g}_2}[I_m\dot{\sigma}(g_{1i}\cos\nu + g_{2i}\sin\nu)A],$$

where in the previous expression

$$A = \frac{\langle \boldsymbol{v}_0, \boldsymbol{v}_\nu \rangle\sigma(g_{1i}\cos\nu + g_{2i}\sin\nu)(g_{2i}\cos\nu - g_{1i}\sin\nu)\psi'(\|\boldsymbol{v}_\nu\|_2)}{\psi(\|\boldsymbol{v}_0\|_2)\psi(\|\boldsymbol{v}_\nu\|_2)^2\|\boldsymbol{v}_\nu\|_2}.$$

By the preceding (product) absolute integrability check when taking first derivatives, we can apply Fubini's theorem to split the integral as we did with $\Xi_1$. We define, with $\boldsymbol{g}_2$ fixed, the function

$$f_2(\nu, g) = B\frac{\langle \boldsymbol{v}_0(\boldsymbol{g}_1), \boldsymbol{v}_\nu(\boldsymbol{g}_1, \hat{\pi}_i(g, \pi_i(\boldsymbol{g}_2))) \rangle}{\psi(\|\boldsymbol{v}_0(\boldsymbol{g}_1)\|_2)\psi(\|\boldsymbol{v}_\nu(\boldsymbol{g}_1, \hat{\pi}_i(g, \pi_i(\boldsymbol{g}_2)))\|_2)^2\|\boldsymbol{v}_\nu(\boldsymbol{g}_1, \hat{\pi}_i(g, \pi_i(\boldsymbol{g}_2)))\|_2}$$

where in the previous expression

$$B = \sigma(g_{1i}\cos\nu + g\sin\nu)(g\cos\nu - g_{1i}\sin\nu)\psi'(\|\boldsymbol{v}_\nu(\boldsymbol{g}_1, \hat{\pi}_i(g, \pi_i(\boldsymbol{g}_2)))\|_2),$$

so that

$$\Xi_2(\nu) = \sum_{i=1}^{n} \mathop{\mathbb{E}}_{(g_{2j}):j\neq i}\left[\mathop{\mathbb{E}}_{g_{2i}}\left[f_2(\nu, g_{2i})\dot{\sigma}(g_{1i}\cos\nu + g_{2i}\sin\nu)\right]\right].$$

Now we check that the hypotheses of Lemma E.27 are satisfied for $f_2$. The continuity argument is identical to that employed for $f_1$, as is the joint absolute integrability property of $f_2$. For absolute continuity, we again use $\psi > 0$, $\psi$ smooth, and Lemma E.21 to obtain the derivative at all but finitely many points of $[0, \pi]$ (by the chain rule and the Leibniz rule) as

$$\begin{aligned}
f_2'(\nu, g) =\ & \frac{\langle\boldsymbol{v}_0, \dot{\boldsymbol{v}}_\nu\rangle\sigma(g_{1i}\cos\nu + g\sin\nu)(g\cos\nu - g_{1i}\sin\nu)\psi'(\|\boldsymbol{v}_\nu\|_2)}{\psi(\|\boldsymbol{v}_0\|_2)\psi(\|\boldsymbol{v}_\nu\|_2)^2\|\boldsymbol{v}_\nu\|_2} \\
& + \frac{\langle\boldsymbol{v}_0, \boldsymbol{v}_\nu\rangle\dot{\sigma}(g_{1i}\cos\nu + g\sin\nu)(g\cos\nu - g_{1i}\sin\nu)^2\psi'(\|\boldsymbol{v}_\nu\|_2)}{\psi(\|\boldsymbol{v}_0\|_2)\psi(\|\boldsymbol{v}_\nu\|_2)^2\|\boldsymbol{v}_\nu\|_2} \\
& - \frac{\langle\boldsymbol{v}_0, \boldsymbol{v}_\nu\rangle\sigma(g_{1i}\cos\nu + g\sin\nu)(g_{1i}\cos\nu + g\sin\nu)\psi'(\|\boldsymbol{v}_\nu\|_2)}{\psi(\|\boldsymbol{v}_0\|_2)\psi(\|\boldsymbol{v}_\nu\|_2)^2\|\boldsymbol{v}_\nu\|_2} \\
& - 2\frac{\langle\boldsymbol{v}_0, \boldsymbol{v}_\nu\rangle\sigma(g_{1i}\cos\nu + g\sin\nu)(g\cos\nu - g_{1i}\sin\nu)\psi'(\|\boldsymbol{v}_\nu\|_2)\langle\boldsymbol{v}_\nu, \dot{\boldsymbol{v}}_\nu\rangle}{\psi(\|\boldsymbol{v}_0\|_2)\psi(\|\boldsymbol{v}_\nu\|_2)^3\|\boldsymbol{v}_\nu\|_2^2} \\
& - \frac{\langle\boldsymbol{v}_0, \boldsymbol{v}_\nu\rangle\sigma(g_{1i}\cos\nu + g\sin\nu)(g\cos\nu - g_{1i}\sin\nu)\psi'(\|\boldsymbol{v}_\nu\|_2)\langle\boldsymbol{v}_\nu, \dot{\boldsymbol{v}}_\nu\rangle}{\psi(\|\boldsymbol{v}_0\|_2)\psi(\|\boldsymbol{v}_\nu\|_2)^2\|\boldsymbol{v}_\nu\|_2^3} \\
& + \frac{\langle\boldsymbol{v}_0, \boldsymbol{v}_\nu\rangle\sigma(g_{1i}\cos\nu + g\sin\nu)(g\cos\nu - g_{1i}\sin\nu)\psi''(\|\boldsymbol{v}_\nu\|_2)\langle\boldsymbol{v}_\nu, \dot{\boldsymbol{v}}_\nu\rangle}{\psi(\|\boldsymbol{v}_0\|_2)\psi(\|\boldsymbol{v}_\nu\|_2)^2\|\boldsymbol{v}_\nu\|_2^2}.
\end{aligned}$$

Because $\psi'$ or $\psi''$ and our convention handle cancellation in the case where $\|\boldsymbol{v}_\nu\|_2 = 0$, we can proceed when necessary with the convenient estimate

$$\left|\frac{\sigma(g_{1i}\cos\nu + g\sin\nu)}{\|\boldsymbol{v}_\nu(\boldsymbol{g}_1, \hat{\pi}_i(g, \pi_i(\boldsymbol{g}_2)))\|_2}\right| \leq 1,$$

which follows from the fact that $\|\boldsymbol{u}\|_\infty \leq \|\boldsymbol{u}\|_2$ for any $\boldsymbol{u} \in \mathbb{R}^n$. As with $\Xi_1$, we then estimate the magnitude of $f_2'$ using Lemma E.31, Cauchy-Schwarz, the triangle inequality, and square-root subadditivity (skipping some steps that we wrote out in the $\Xi_1$ estimate):

$$\begin{aligned}
|f_2'(\nu, g)| &\leq 64C(|g| + |g_{1i}|)\|\boldsymbol{v}_0\|_2\left(\|\dot{\boldsymbol{v}}_\nu\|_2\left(2 + \frac{C'}{C}\right) + (|g| + |g_{1i}|)(2 + 8\|\dot{\boldsymbol{v}}_\nu\|_2)\right) \\
&\leq 64C(|g| + |g_{1i}|)\|\boldsymbol{g}_1\|_2\left(\begin{array}{c}(\|\boldsymbol{g}_1\|_2 + \|\pi_i(\boldsymbol{g}_2)\|_2 + |g|)\left(2 + \frac{C'}{C}\right) \\ +(|g| + |g_{1i}|)(2 + 8(\|\boldsymbol{g}_1\|_2 + \|\pi_i(\boldsymbol{g}_2)\|_2 + |g|))\end{array}\right),
\end{aligned}$$
(E.45)

which is jointly integrable over $[0, \pi] \times \mathbb{R}$, and moreover over $[0, \pi] \times \mathbb{R}^n$. We conclude absolute continuity of $f_2(\cdot, g)$ and the integrability property of $f_2'$. For the growth estimate, we argue similarly to our bound on $f_2'$ to get

$$\begin{aligned}
|f_2(\nu, g)| &\leq 64C\|\boldsymbol{v}_0\|_2(|g| + |g_{1i}|)^2 \\
&\leq 64C\|\boldsymbol{g}_1\|_2\left(|g|^2 + 2|g_{1i}||g| + |g_{1i}|^2\right);
\end{aligned}$$
(E.46)

the RHS in the final inequality is a quadratic function of $|g|$, and we therefore obtain a suitable growth estimate with $p = 2$ and $C' = 64C\|\boldsymbol{g}_1\|(1 + 2|g_{1i}| + |g_{1i}|^2)$ as soon as $|g| \geq 1$. We can therefore apply Lemma E.27 to get that for all but a negligible set of $\boldsymbol{g}_1$ that

$$\Xi_2(\nu) = \sum_{i=1}^{n} \mathop{\mathbb{E}}_{(g_{2j}):j\neq i}\left[\mathop{\mathbb{E}}_{g_{2i}}\left[f_2(0, g_{2i})\dot{\sigma}(g_{1i})\right] + \int_0^\nu\left(\begin{array}{c}\mathbb{E}_{g_{2i}}[f_2'(t, g_{2i})\dot{\sigma}(g_{1i}\cos t + g_{2i}\sin t)] \\ -g_{1i}\frac{f_2(t, -g_{1i}\cot t)\rho(-g_{1i}\cot t)}{\sin^2 t}\end{array}\right)\,\mathrm{d}t\right].$$

The estimates (E.45) and (E.46) show, respectively, that $f_2'$ and $f_2$ are absolutely integrable functions of $(\nu, \boldsymbol{g}_2)$. Because $\sigma(g_{1i}\cos\nu - g_{1i}\cot\nu\sin\nu) = 0$, we have (fortuitously)

$$f_2(t, -g_{1i}\cot t) = 0,$$

so that there is no Dirac term in the derivative expression for $\Xi_2$. An application of Fubini's theorem then allows us to re-combine the split integrals in the previous expression:

$$\Xi_2(\nu) = \sum_{i=1}^{n} \mathop{\mathbb{E}}_{\boldsymbol{g}_2}[f_2(0, g_{2i})\dot{\sigma}(g_{1i})] + \int_0^{\nu} \left( \mathop{\mathbb{E}}_{\boldsymbol{g}_2}[f_2'(t, g_{2i})\dot{\sigma}(g_{1i}\cos t + g_{2i}\sin t)] \right) \mathrm{d}t.$$

We have by linearity of the integral

$$\sum_{i=1}^{n} \mathop{\mathbb{E}}_{\boldsymbol{g}_2}[f_2(0, g_{2i})\dot{\sigma}(g_{1i})] = \mathop{\mathbb{E}}_{\boldsymbol{g}_2}\left[ \frac{\langle \boldsymbol{v}_0, \dot{\boldsymbol{v}}_0\rangle \|\boldsymbol{v}_0\| \psi'(\|\boldsymbol{v}_0\|)}{\psi(\|\boldsymbol{v}_0\|)^3} \right] = 0,$$

where the last equality applied independence of $\boldsymbol{g}_1$ and $\boldsymbol{g}_2$, as in the zero-order term of $\Xi_1$. Finally, we apply linearity of the integral to move the summation over $i$ back inside the remaining integrals, obtaining

$$\begin{aligned}
\Xi_2(\nu) = \int_0^{\nu} \Bigg( &\mathop{\mathbb{E}}_{\boldsymbol{g}_2}\left[ \frac{\langle \boldsymbol{v}_0, \dot{\boldsymbol{v}}_t\rangle\langle \boldsymbol{v}_t, \dot{\boldsymbol{v}}_t\rangle \psi'(\|\boldsymbol{v}_t\|_2)}{\psi(\|\boldsymbol{v}_0\|_2)\psi(\|\boldsymbol{v}_t\|_2)^2\|\boldsymbol{v}_t\|_2} + \frac{\langle \boldsymbol{v}_0, \boldsymbol{v}_t\rangle\|\dot{\boldsymbol{v}}_t\|_2^2\psi'(\|\boldsymbol{v}_t\|_2)}{\psi(\|\boldsymbol{v}_0\|_2)\psi(\|\boldsymbol{v}_t\|_2)^2\|\boldsymbol{v}_t\|_2} \right] \\
&+ \mathop{\mathbb{E}}_{\boldsymbol{g}_2}\left[ -\frac{\langle \boldsymbol{v}_0, \boldsymbol{v}_t\rangle\psi'(\|\boldsymbol{v}_t\|_2)\|\boldsymbol{v}_t\|_2}{\psi(\|\boldsymbol{v}_0\|_2)\psi(\|\boldsymbol{v}_t\|_2)^2} - 2\frac{\langle \boldsymbol{v}_0, \boldsymbol{v}_t\rangle\langle \boldsymbol{v}_t, \dot{\boldsymbol{v}}_t\rangle^2\psi'(\|\boldsymbol{v}_t\|_2)}{\psi(\|\boldsymbol{v}_0\|_2)\psi(\|\boldsymbol{v}_t\|_2)^3\|\boldsymbol{v}_t\|_2^2} \right] \\
&+ \mathop{\mathbb{E}}_{\boldsymbol{g}_2}\left[ -\frac{\langle \boldsymbol{v}_0, \boldsymbol{v}_t\rangle\langle \boldsymbol{v}_t, \dot{\boldsymbol{v}}_t\rangle^2\psi'(\|\boldsymbol{v}_t\|_2)}{\psi(\|\boldsymbol{v}_0\|_2)\psi(\|\boldsymbol{v}_t\|_2)^2\|\boldsymbol{v}_t\|_2^3} + \frac{\langle \boldsymbol{v}_0, \boldsymbol{v}_t\rangle\langle \boldsymbol{v}_t, \dot{\boldsymbol{v}}_t\rangle^2\psi''(\|\boldsymbol{v}_t\|_2)}{\psi(\|\boldsymbol{v}_0\|_2)\psi(\|\boldsymbol{v}_t\|_2)^2\|\boldsymbol{v}_t\|_2^2} \right] \Bigg) \mathrm{d}t.
\end{aligned}$$

Since $f(\nu, \boldsymbol{g}_1) = f(0, \boldsymbol{g}_1) + \int_0^{\nu} \mathbb{E}_{\boldsymbol{g}_2}[\Xi_1(t) - \Xi_2(t)]\,\mathrm{d}t$, the claim follows. $\qquad\square$

**Lemma E.27.** *Let $\mu$ denote the distribution of a $\mathcal{N}(0, (2/n))$ random variable, and let $\rho$ denote its density. Let $u \in \mathbb{R}$ and $u \neq 0$, and let $f : [0, \pi] \times \mathbb{R} \to \mathbb{R}$ satisfy:*

1. *$f$ is continuous in its second argument with its first argument fixed;*

2. *$f$ is absolutely continuous in its first argument with its second argument fixed, with a.e. derivative $f'$;*

3. *$f$ and $f'$ are absolutely integrable with respect to the product of Lebesgue measure and $\mu$ over $[0, \pi] \times \mathbb{R}$;*

4. *There exist $p \geq 1$ and $C > 0$ constants independent of $x$ such that $|f(\nu, x)| \leq C|x|^p$ whenever $|x| \geq 1$.*

*Consider the function*

$$q(\nu) = \int_{\mathbb{R}} f(\nu, x)\dot{\sigma}(u\cos\nu + x\sin\nu)\,\mathrm{d}\mu(x).$$

*Then $q$ is absolutely continuous, and the following first-order Taylor expansion holds:*

$$q(\nu) = q(0) + \int_0^{\nu} \left( -u\frac{f(t, -u\cot t)\,\rho(-u\cot t)}{\sin^2 t} + \int_{\mathbb{R}} f'(t, x)\dot{\sigma}(u\cos t + x\sin t)\,\mathrm{d}\mu(x) \right) \mathrm{d}t.$$

*Proof.* For $m \in \mathbb{N}$, define

$$\dot{\sigma}_m(x) = \begin{cases} 0 & x \leq 0 \\ mx & 0 \leq x \leq m^{-1} \\ 1 & x \geq m^{-1}. \end{cases}$$

Then $0 \leq \dot{\sigma}_m \leq 1$; $\dot{\sigma}_m$ is continuous, hence Borel measurable; $\dot{\sigma}_m \to \dot{\sigma}$ pointwise as $m \to \infty$; and $\dot{\sigma}_m$ is differentiable on $\mathbb{R}$ except at $x \in \{0, m^{-1}\}$, with derivative $\ddot{\sigma}_m = m\mathbb{1}_{0 \leq x \leq m^{-1}}$. Moreover, we have

$$\int_{\mathbb{R}} m\mathbb{1}_{0 \leq x \leq m^{-1}}\,\mathrm{d}x = 1,$$

and the first-order Taylor expansion

$$\dot{\sigma}_m(x) = \int_0^x m\mathbb{1}_{0 \leq x' \leq m^{-1}}\,\mathrm{d}x'.$$

Define

$$q_m(\nu) = \int_{\mathbb{R}} f(\nu, x)\dot{\sigma}_m(u\cos\nu + x\sin\nu)\,\mathrm{d}\mu(x).$$

Then at every $\nu \in [0, \pi]$, we have by assumption

$$\int_{\mathbb{R}} |f(\nu, x)\dot{\sigma}_m(u\cos\nu + x\sin\nu)|\,\mathrm{d}\mu(x) \le \int_{\mathbb{R}} |f(\nu, x)|\,\mathrm{d}\mu(x) < +\infty,$$

so that the dominated convergence theorem implies

$$\lim_{m\to\infty} q_m(\nu) = q(\nu).$$

By the chain rule, the expression $\dot{\sigma}_m(x) = -\max\{-m\max\{x, 0\}, -1\}$, and Lemma E.21, $\nu \mapsto \dot{\sigma}_m(u\cos\nu + x\sin\nu)$ is an absolutely continuous function of $\nu \in [0, \pi]$, and we therefore have by the product rule for AC functions on an interval (Cohn, 2013, Corollary 6.3.9)

$$q_m(\nu) = q_m(0)$$
$$+ \int_{\mathbb{R}} \mathrm{d}\mu(x) \int_0^\nu \mathrm{d}t \bigg( f'(t, x)\dot{\sigma}_m(u\cos t + x\sin t)$$
$$+ mf(t, x)(x\cos t - u\sin t)\mathbb{1}_{0 \le u\cos\nu + x\sin\nu \le m^{-1}} \bigg).$$

We have

$$\int_{\mathbb{R}} \int_0^\pi |f'(t, x)\dot{\sigma}_m(u\cos t + x\sin t)|\,\mathrm{d}t\,\mathrm{d}\mu(x) \le \int_{\mathbb{R}} \int_0^\pi |f'(t, x)|\,\mathrm{d}t\,\mathrm{d}\mu(x) < +\infty,$$

by assumption, and

$$\int_{\mathbb{R}} \left| f(t, x)(x\cos t - u\sin t)\mathbb{1}_{0 \le u\cos t + x\sin t \le m^{-1}} \right| \mathrm{d}\mu(x)$$
$$\le \left( \int_{\mathbb{R}} f(t, x)^2\,\mathrm{d}\mu(x) \right)^{1/2} \left( \int_{\mathbb{R}} (x\cos t - u\sin t)^2\,\mathrm{d}\mu(x) \right)^{1/2}$$
$$\le C_f \left( |u| + \left( \int_{\mathbb{R}} x^2\,\mathrm{d}\mu(x) \right)^{1/2} \right) < +\infty,$$

by the growth assumption on $f$ and the Schwarz inequality. Applying compactness of $[0, \pi]$ and the lack of $\nu$ dependence in the final inequality above, an application of Fubini's theorem therefore yields

$$q_m(\nu) = q_m(0)$$
$$+ \int_0^\nu \int_{\mathbb{R}} \mathrm{d}\mu(x)\,\mathrm{d}t \bigg( f'(t, x)\dot{\sigma}_m(u\cos t + x\sin t)$$
$$+ mf(t, x)(x\cos t - u\sin t)\mathbb{1}_{0 \le u\cos\nu + x\sin\nu \le m^{-1}} \bigg).$$

By dominated convergence and the first of the preceding two product integrability checks, it is clear

$$\lim_{m\to\infty} \int_0^\nu \int_{\mathbb{R}} f'(t, x)\dot{\sigma}_m(u\cos t + x\sin t)\,\mathrm{d}\mu(x)\,\mathrm{d}t = \int_0^\nu \int_{\mathbb{R}} f'(t, x)\dot{\sigma}(u\cos t + x\sin t)\,\mathrm{d}\mu(x)\,\mathrm{d}t.$$

For the second term, we need to proceed more carefully. For $k \in \mathbb{N}$ sufficiently large for the integral to be over a nonempty interval, we consider

$$q_{m,k}(\nu) := \int_{k^{-1}}^{\nu - k^{-1}} \mathrm{d}t \int_{\mathbb{R}} mf(t, x)(x\cos t - u\sin t)\frac{1}{\sqrt{2\pi c^2}}e^{-\frac{x^2}{2c^2}}\mathbb{1}_{0 \le u\cos t + x\sin t \le m^{-1}}\,\mathrm{d}x,$$

which is a truncated version of the integral constituting the second term in $q_m$, with a change of variables applied to explicitly show the density corresponding to $\mu$, and where we write $c^2 = 2/n$.

In particular, by the calculation used to apply Fubini's theorem in this context previously, we have by dominated convergence

$$\lim_{k \to \infty} q_{m,k}(\nu) = \int_0^\nu \int_{\mathbb{R}} m f(t,x)(x \cos t - u \sin t) \mathbb{1}_{0 \le u \cos t + x \sin t \le m^{-1}} \, d\mu(x) \, dt.$$

By the product integrability assumption on $f$ and Fubini's theorem, we can consider the inner $\mathbb{R}$-integral for fixed $t$, and due to our truncation we have $0 < t < \pi$; we therefore change variables $x \mapsto x \sin^{-1} t$ in the inner integral to get

$$q_{m,k}(\nu) = \int_{k^{-1}}^{\nu - k^{-1}} dt \int_{\mathbb{R}} m f\left(t, \frac{x}{\sin t}\right) \left(x \frac{\cos t}{\sin^2 t} - u\right) \frac{1}{\sqrt{2\pi c^2}} e^{-\frac{x^2}{2c^2 \sin^2 t}} \mathbb{1}_{0 \le u \cos t + x \le m^{-1}} \, dx.$$

If $0 < t < \pi$ and $x \in \mathbb{R}$, define

$$g(t,x) = f\left(t, \frac{x}{\sin t}\right) \left(x \frac{\cos t}{\sin^2 t} - u\right) \frac{1}{\sqrt{2\pi c^2}} e^{-\frac{x^2}{2c^2 \sin^2 t}},$$

so that, after an additional change of variables $x \mapsto x - u \cos t$, we obtain

$$q_{m,k}(\nu) = m \int_{k^{-1}}^{\nu - k^{-1}} dt \int_{\mathbb{R}} g\left(t, x - u \cos t\right) \mathbb{1}_{0 \le x \le m^{-1}} \, dx.$$

Using the growth estimate for $f$, we have

$$|g(t, x - u \cos t)| \le C \frac{|x - u \cos t|^p |x \cos t - u|}{\sin^{p+2} t} \exp\left(-\frac{(x - u \cos t)^2}{2c^2 \sin^2 t}\right),$$

where $C > 0$ depends only on $c$. We are going to bound this quantity under the assumption that $|x| \le |u|/2$, where we use the assumption $|u| > 0$. First, note that when $\pi/4 \le t \le 3\pi/4$, we have $\sin t \ge 1/\sqrt{2}$, and we always have $\sin t \le 1$ for $0 \le t \le \pi$; so in this regime

$$|g(t, x - u \cos t)| \le C 2^{p/2+1} |x - u \cos t|^p |x \cos t - u| \exp\left(-\frac{(x - u \cos t)^2}{2c^2}\right),$$

which is a continuous function of $(t, x)$, and is therefore bounded by a constant depending only on $c, f, u$ over the compact set $[\pi/4, 3\pi/4] \times [-u/2, u/2]$. Next, we consider the case $0 < t \le \pi/4$; by the symmetry $\sin(\pi - t) = \sin t$, controlling $|g(t, x - u \cos t)|$ in this regime implies control of it in the regime $3\pi/4 \le t < \pi$. Here, we note that by our assumption on $t$ and the triangle inequality

$$|x - u \cos t| \ge |u||\cos t| - |x|$$
$$\ge |u|(|\cos t| - \tfrac{1}{2}) \ge K|u|,$$

where we can take $K = 2^{-1/2} - 2^{-1} > 0$. Applying the triangle inequality and the condition on $|x|$ gives

$$|g(t, x - u \cos t)| \le C(3/2)^{p+1} \frac{|u|^{p+1}}{\sin^{p+2} t} \exp\left(-\frac{K^2 u^2}{2c^2 \sin^2 t}\right),$$

which only depends on $t$. For any constants $c', C' > 0$, the continuous map $y \mapsto C|y|^{p+2} e^{-c' y^2}$ is a bounded function of $y \in \mathbb{R}$ by L'Hôpital's rule applied to determine $\lim_{y \to \pm\infty} |y|^p e^{-y^2} = 0$ for any $p > 0$. It follows that there is a constant $M \ge 0$ depending only on $c, u, p$ such that $|g(t, x - u \cos t)| \le M$ whenever $0 \le t\pi/4$; we obtain the result for $t = 0$ by the previous limit calculation. Applying symmetry and taking the sum of our two bounds then yields $|g(t, x - u \cos t)| \le M'$ for $M' \ge 0$ not depending on $k, m$ whenever $(t, x) \in [0, \pi] \times [-u/2, u/2]$.

Now, we have after one additional change of variables $x \mapsto x m^{-1}$

$$q_{m,k}(\nu) = \int_{k^{-1}}^{\nu - k^{-1}} dt \int_{\mathbb{R}} g\left(t, x m^{-1} - u \cos t\right) \mathbb{1}_{0 \le x \le 1} \, dx.$$

We can invoke our $M'$ bound when $x m^{-1} \le |u|/2$, and the indicator enforces $|x| \le 1$; thus, taking $m \ge 2/|u|$ (here we use $|u| > 0$ critically) implies

$$\int_{k^{-1}}^{\nu - k^{-1}} dt \int_{\mathbb{R}} |g\left(t, x m^{-1} - u \cos t\right) \mathbb{1}_{0 \le x \le 1} \, dx| \le M' \int_{k^{-1}}^{\nu - k^{-1}} dt < +\infty,$$

so that by dominated convergence, we have

$$\lim_{k\to\infty} q_{m,k}(\nu) = \int_0^\nu \mathrm{d}t \int_{\mathbb{R}} g\left(t, xm^{-1} - u\cos t\right) \mathbb{1}_{0 \le x \le 1} \, \mathrm{d}x.$$

By the same estimate together with second-argument continuity of $f$, hence of $g$, we have by the dominated convergence theorem

$$\lim_{m\to\infty} \lim_{k\to\infty} q_{m,k}(\nu) = \int_0^\nu g(t, -u\cos t)\, \mathrm{d}t = -u \int_0^\nu \frac{f\left(t, -u\cot t\right)}{\sin^2 t} \frac{1}{\sqrt{2\pi c^2}} e^{-\frac{u^2 \cot^2 t}{2c^2}} \, \mathrm{d}t.$$

Combining with our results on $q_m$ and the first term, we conclude

$$
\begin{aligned}
q(\nu) = q(0) \\
+ \int_0^\nu \mathrm{d}t \left( -u \frac{f\left(t, -u\cot t\right)}{\sin^2 t} \frac{1}{\sqrt{2\pi c^2}} e^{-\frac{u^2 \cot^2 t}{2c^2}} + \int_{\mathbb{R}} f'(t, x)\dot{\sigma}(u\cos t + x\sin t)\, \mathrm{d}\mu(x) \right),
\end{aligned}
$$

as claimed. $\qquad\square$

### E.3.6 MISCELLANEOUS ANALYTICAL RESULTS

**Lemma E.28.** *If $m > 0$, then $\bar{\varphi}$ is 1-Lipschitz.*

*Proof.* We recall

$$\bar{\varphi}(\nu) = \mathop{\mathbb{E}}_{\boldsymbol{g}_1, \boldsymbol{g}_2 \sim \text{i.i.d.} \, \mathcal{N}(\mathbf{0}, (2/n)\boldsymbol{I})} \left[\cos^{-1} X_\nu\right].$$

Considering instead the related function $\tilde{\varphi}$ defined by

$$\tilde{\varphi}(\nu) = \mathop{\mathbb{E}}_{\boldsymbol{g}_1, \boldsymbol{g}_2 \sim \text{i.i.d.} \, \mathcal{N}(\mathbf{0}, (2/n)\boldsymbol{I})} [\mathbb{1}_{\mathcal{E}_1} \phi(\nu, \boldsymbol{g}_1, \boldsymbol{g}_2)],$$

where

$$\phi(\nu, \boldsymbol{g}_1, \boldsymbol{g}_2) = \cos^{-1} \left( \frac{\langle \boldsymbol{v}_0, \boldsymbol{v}_\nu \rangle}{\|\boldsymbol{v}_0\|_2 \|\boldsymbol{v}_\nu\|_2} \right),$$

we notice

$$\bar{\varphi}(\nu) = \tilde{\varphi}(\nu) + (\pi/2)\mu(\mathcal{E}_1^{\mathsf{c}}).$$

It is therefore equivalent to show that $\tilde{\varphi}$ is 1-Lipschitz; but this follows from Lemma E.22. $\qquad\square$

**Lemma E.29** (Even Moments). *If $k \in \mathbb{N}$ and $k \le n$, one has*

$$\left| \mathbb{E}\left[\|\boldsymbol{v}_\nu\|_2^{2k}\right] - 1 \right| \le C_k n^{-1}, \qquad \left| \mathbb{E}\left[\|\dot{\boldsymbol{v}}_\nu\|_2^{2k}\right] - 1 \right| \le C_k n^{-1},$$

*where $C_k \le (k-1)^2 4^{k-1}(2k-1)!!$.*

*Proof.* First notice that the claim is immediate if $k = 1$, since $\mathbb{E}\left[\|\boldsymbol{v}_\nu\|_2^2\right] = 1$. We therefore proceed assuming $k > 1$. Also notice that Lemmas G.11 and E.17 show that $\dot{\boldsymbol{v}}_\nu$ and $\boldsymbol{v}_\nu$ have matching even moments, so it suffices to prove the claim for $\boldsymbol{v}_\nu$. By rotational invariance, we can write

$$
\begin{aligned}
\mathbb{E}\left[\|\boldsymbol{v}_\nu\|_2^{2k}\right] &= \frac{2^k}{n^k} \mathop{\mathbb{E}}_{g_i \sim \mathcal{N}(0,1)} \left[ \left( \sum_{i=1}^n \sigma(g_i)^2 \right)^k \right] \\
&= \frac{2^k}{n^k} \sum_{1 \le i_1, \dots, i_k \le n} \mathbb{E}\left[ \prod_{j=1}^k \sigma(g_{i_j})^2 \right],
\end{aligned}
$$

where the last sum is taken over all elements of $[n]^k$. We split this sum into a sum over terms whose expectations contain no repeated indices, and a sum over all other terms. There are exactly $k!\binom{n}{k}$ ways to choose a $k$-multi-index from an alphabet of size $n$ without repetitions—select the $k$ distinct

indices, then arrange them in every possible way—and multi-indices without repetitions correspond to terms in the sum where the expectation factors completely, by independence, so we can write

$$\mathbb{E}\big[\|\boldsymbol{v}_\nu\|_2^{2k}\big] = \frac{2^k}{n^k}\left(k!\binom{n}{k}\mathbb{E}\big[\sigma(g_1)^2\big]^k + \sum_{\substack{1\leq i_1,\ldots,i_k\leq n \\ \text{only repeated indices}}}\mathbb{E}\left[\prod_{j=1}^k\sigma(g_{i_j})^2\right]\right).$$

We will prove the elementary estimate

$$\left|n^k - k!\binom{n}{k}\right| \leq (k-1)^2 n^{k-1}2^{k-2}. \tag{E.47}$$

Assuming it for the time being, we use that $\mathbb{E}\big[\sigma(g_1)^2\big]^k = 2^{-k}$ to conclude

$$\left|(2/n)^k k!\binom{n}{k}\mathbb{E}\big[\sigma(g_1)^2\big]^k - 1\right| \leq (k-1)^2 2^{k-2}n^{-1}.$$

Next we study the expectation-of-products arising in the sum. The expectation factors over distinct indices; we can classify repeated indices in a multi-index by partitions $j_1+\ldots j_m = k$, where each $j_l$ is a positive integer. Formally, for each multi-index $(i_1,\ldots,i_k)$, there is a partition $j_1 + \ldots j_m = k$ such that

$$\mathbb{E}\left[\prod_{j=1}^k\sigma(g_{i_j})^2\right] = \prod_{l=1}^m\mathbb{E}\big[\sigma(g_{i_{p(l)}})^{2j_l}\big],$$

where $p : [m] \to [k]$ is injective. We can evaluate these expectations using the result $\mathbb{E}\big[\sigma(g_1)^{2k}\big] = \frac{1}{2}(2k-1)!!$, because the coordinates of $\boldsymbol{g}$ are i.i.d.:

$$\prod_{l=1}^m\mathbb{E}\big[\sigma(g_{i_{p(l)}})^{2j_l}\big] = \frac{1}{2^m}\prod_{i=1}^m(2j_i-1)!!.$$

We claim that

$$\frac{1}{2^m}\prod_{i=1}^m(2j_i-1)!! \leq \frac{1}{2}(2k-1)!!, \tag{E.48}$$

which is the expectation obtained from a term with all indices equal, whence

$$\left|\frac{2^k}{n^k}\sum_{\substack{1\leq i_1,\ldots,i_k\leq n \\ \text{only repeated indices}}}\mathbb{E}\left[\prod_{j=1}^k\sigma(g_{i_j})^2\right]\right| \leq \frac{2^k}{n^k}n^{k-1}(k-1)^2 2^{k-2}\mathbb{E}\big[\sigma(g_1)^{2k}\big]$$

$$= \big((k-1)^2 2^{2k-3}(2k-1)!!\big)n^{-1}$$

by (E.47), which gives a bound on the number of terms in the sum. Noticing that this constant is larger than $(k-1)^2 2^{k-2}$, we can conclude the claimed estimate on $C_k$ provided we can justify (E.48). For this, it suffices to show

$$1 \leq 2^{m-1}\frac{(2k-1)!!}{\prod_{i=1}^m(2j_i-1)!!}.$$

Observe that $m \geq 1$ for any partition, so $2^{m-1} \geq 1$ and we need only study the second term on the righthand side. We write this term as

$$\frac{(2k-1)!!}{\prod_{i=1}^m(2j_i-1)!!} = \frac{\prod_{i=1}^k(2i-1)}{\prod_{i=1}^m\prod_{l=1}^{j_i}(2l-1)}.$$

The fact that $j_1 + \cdots + j_m = k$ implies that there are $k$ factors in the denominator, so we can put the factors in the numerator and denominator into one-to-one correspondence. Consider the ordering of the factors in the denominator $(\prod_{l=1}^{j_1}(2l-1))\ldots(\prod_{l=1}^{j_m}(2l-1))$. Then

$$\frac{\prod_{i=1}^k(2i-1)}{\prod_{i=1}^{j_1}(2i-1)} = \prod_{i=j_1+1}^k(2i-1).$$

If $j_1 = k$, then this product is empty and $m = 1$, so the claim is established. If not, then we proceed to the next group of factors in the denominator: we get

$$\frac{\prod_{i=j_1+1}^{k}(2i-1)}{\prod_{i=1}^{j_2}(2i-1)} \geq 1,$$

because $j_1 > 0$ implies that every term in the numerator (ordered in ascending order) is larger than the corresponding term in the denominator. This gives the claim in the case $m = 2$; for $m > 2$, we conclude the claim by induction.

To close the loop, we prove (E.47). Using simple algebra, we observe

$$n^k - k!\binom{n}{k} = n^k - n(n-1)\ldots(n-k+1)$$

$$= n^k\left(1 - \prod_{j=1}^{k-1}\left(1 - \frac{j}{n}\right)\right),$$

and we note bounds

$$\left(1 - \frac{k-1}{n}\right)^{k-1} \leq \prod_{j=1}^{k-1}\left(1 - \frac{j}{n}\right) \leq 1.$$

Working on the upper bound first, we obtain with the help of the binomial theorem

$$1 - \prod_{j=1}^{k-1}\left(1 - \frac{j}{n}\right) \leq 1 - \left(1 - \frac{k-1}{n}\right)^{k-1}$$

$$= \sum_{j=1}^{k-1}\binom{k-1}{j}(-1)^{j+1}\left(\frac{k-1}{n}\right)^j$$

$$\leq \frac{k-1}{n}\sum_{j=0}^{k-2}\binom{k-1}{j+1}\left(\frac{k-1}{n}\right)^j,$$

where the last expression removes cancellation by making each term in the sum nonnegative, then applies a change of index. With the identity $\binom{k-1}{j+1} = (k-1)/(j+1)\binom{k-2}{j}$, we proceed as

$$\frac{k-1}{n}\sum_{j=0}^{k-2}\binom{k-1}{j+1}\left(\frac{k-1}{n}\right)^j = \frac{(k-1)^2}{n}\sum_{j=0}^{k-2}\binom{k-2}{j}\frac{1}{j+1}\left(\frac{k-1}{n}\right)^j$$

$$\leq \frac{(k-1)^2}{n}\sum_{j=0}^{k-2}\binom{k-2}{j}\left(\frac{k-1}{n}\right)^j$$

$$= \frac{(k-1)^2}{n}\left(1 + \frac{k-1}{n}\right)^{k-2},$$

given that $1/(j+1) \leq 1$. Since $n \geq k$, this gives

$$n^k - k!\binom{n}{k} \leq (k-1)^2 n^{k-1} 2^{k-2}.$$

The upper bound on the product gives immediately $n^k - k!\binom{n}{k} \geq 0$, which completes the proof.

$\square$

**Lemma E.30** (Mixed Moments). *Let $g^1, \ldots, g^n$ denote the $n$ (i.i.d. according to $\mathcal{N}(\mathbf{0}, (2/n)\mathbf{I}_2)$) rows of the matrix $\mathbf{G}$. Let $k \in [n]$, and for each $1 \leq j \leq k$ let $f_j : \mathbb{R}^2 \to \mathbb{R}$ be a function such that*

1. $\mathbb{E}[|f_j(g^1)|^p]^{1/p} \leq Cn^{-1}p$, *with $C > 0$ an absolute constant and $p \geq 1$;*

2. $\mathbb{E}[|f_j(g^1)|] \leq n^{-1}$.

*Consider the quantities*

$$A = \mathbb{E}\left[\prod_{j=1}^{k}\left(\sum_{i=1}^{n} f_j(\boldsymbol{g}^i)\right)\right]; \qquad B = n^k \prod_{j=1}^{k} \mathbb{E}\left[f_j(\boldsymbol{g}^1)\right].$$

*Then one has $|A - B| \leq Cn^{-1}$, with the constant depending only on $k$.*

*Proof.* Start by writing

$$
\begin{aligned}
A &= \sum_{1 \leq i_1,\ldots,i_k \leq n} \mathbb{E}\left[\prod_{j=1}^{k} f_j(\boldsymbol{g}^{i_j})\right] \\
&= k!\binom{n}{k}\prod_{j=1}^{k}\mathbb{E}\left[f_j(\boldsymbol{g}^1)\right] + \sum_{\substack{1 \leq i_1,\ldots,i_k \leq n \\ \text{only repeated indices}}} \mathbb{E}\left[\prod_{j=1}^{k} f_j(\boldsymbol{g}^{i_j})\right] \\
&= n^{-k}k!\binom{n}{k}B + \sum_{\substack{1 \leq i_1,\ldots,i_k \leq n \\ \text{only repeated indices}}} \mathbb{E}\left[\prod_{j=1}^{k} f_j(\boldsymbol{g}^{i_j})\right]
\end{aligned}
$$

as in Lemma E.29. Applying the triangle inequality and the first moment assumption on the functions $f_j$, we get

$$\left|n^{-k}k!\binom{n}{k}B - B\right| = |B|\left|k!\binom{n}{k}n^{-k} - 1\right| \leq (k-1)^2 2^{k-2} n^{-1},$$

with the last inequality following from the estimate (E.47). For the remaining term, we have by the triangle inequality

$$\left|\sum_{\substack{1 \leq i_1,\ldots,i_k \leq n \\ \text{only repeated indices}}} \mathbb{E}\left[\prod_{j=1}^{k} f_j(\boldsymbol{g}^{i_j})\right]\right| \leq \left|n^k - k!\binom{n}{k}\right| \sup_{(i_1,\ldots,i_k) \subset [n]^k} \left|\mathbb{E}\left[\prod_{j=1}^{k} f_j(\boldsymbol{g}^{i_j})\right]\right|$$

$$\leq (k-1)^2 n^{k-1} 2^{k-2} \sup_{(i_1,\ldots,i_k) \subset [n]^k} \left|\mathbb{E}\left[\prod_{j=1}^{k} f_j(\boldsymbol{g}^{i_j})\right]\right|,$$

using again (E.47) to control the number of terms in the sum. To control the supremum, we apply the Schwarz inequality $k - 1$ times to get

$$
\begin{aligned}
\left|\mathbb{E}\left[\prod_{j=1}^{k} f_j(\boldsymbol{g}^{i_j})\right]\right| &\leq \mathbb{E}\left[f_1(\boldsymbol{g}^{i_1})^2\right]^{1/2}\mathbb{E}\left[\prod_{j=2}^{k} f_j(\boldsymbol{g}^{i_j})^2\right]^{1/2} \\
&\leq \ldots \\
&\leq \left(\prod_{j=1}^{k-1}\mathbb{E}\left[f_j(\boldsymbol{g}^{i_j})^{2^j}\right]^{2^{-j}}\right)\mathbb{E}\left[f_k(\boldsymbol{g}^{i_k})^{2^{k-1}}\right]^{2^{-(k-1)}}.
\end{aligned}
$$

By the subexponential assumption on the functions $f_j$, we have moment growth control, and we therefore have a bound

$$
\begin{aligned}
\left|\mathbb{E}\left[\prod_{j=1}^{k} f_j(\boldsymbol{g}^{i_j})\right]\right| &\leq \left(\prod_{j=1}^{k-1} C_1 n^{-1} 2^j\right) C_1 n^{-1} 2^{k-1} \\
&= C_1^k n^{-k} 2^{(k-1) + \sum_{j=1}^{k-1} j} = C_1^k n^{-k} 2^{\frac{1}{2}(k-1)(k+2)},
\end{aligned}
$$

and consequently

$$\left| \sum_{\substack{1 \le i_1, \ldots, i_k \le n \\ \text{only repeated indices}}} \mathbb{E} \left[ \prod_{j=1}^{k} f_j(\boldsymbol{g}^{i_j}) \right] \right| \le C_1^k (k-1)^2 2^{\frac{1}{2}k(k+3)} n^{-1},$$

which proves the claim. □

**Lemma E.31.** *For any $0 < c \le \frac{1}{2}$, there exists a smooth function $\psi_c : \mathbb{R} \to \mathbb{R}$ satisfying*

1. *$\psi_c(x) = x$ if $x \ge 2c$ and $\psi_c(x) = c$ if $x \le c$, and $\psi_c$ is between $c$ and $2c$ if $c \le x \le 2c$;*

2. *$\psi_c(x) \ge \frac{1}{2}x$;*

3. *There are constants $M_1, M_2 > 0$ depending only on $c$ such that $|\psi_c'| \le M_1$ and $|\psi_c''| \le M_2$.*

*Proof.* The function $f(x) = \mathbb{1}_{x>0} e^{-\frac{1}{x}}$ is smooth on $\mathbb{R}$, and satisfies $0 \le f \le 1$ and $f = 0$ if $x \le 0$. The function

$$\phi_c(x) = \frac{f(x)}{f(x) + f(c - x)}$$

is therefore smooth, satisfies $0 \le \phi_c \le 1$, and satisfies $\phi_c(x) = 0$ if $x \le 0$ and $\phi_c(x) = 1$ if $x \ge c$. Simplifying using the definitions, we can write

$$\phi_c(x) = \begin{cases} 0 & x \le 0 \\ \frac{1}{1 + \exp\left(\frac{c - 2x}{x(c-x)}\right)} & 0 < x < c \\ 1 & x \ge c. \end{cases}$$

It follows that $x \mapsto x\phi_c(x)$ is zero when $x \le 0$, $x$ when $x \ge c$, and in between otherwise. Thus, the function $\psi_c(x) = c + (x - c)\phi_c(x - c)$ satisfies property 1.

For property 2, we note that $\psi_c(x) = c + (x - c)\phi_c(x - c)$ implies that $\psi_c \ge c$, since $\phi_c(x - c) = 0$ whenever $x \le c$ and $\phi_c \ge 0$. Since $\psi_c(x) = x$ when $x \ge 2c$, we can then conclude $\psi_c(x) \ge \frac{1}{2}x$, since $\frac{1}{2}x \le c$ when $x \le 2c$ and $\frac{1}{2}x \le x$ when $x \ge 2c$.

For property 3, we note that by property 1, $\psi_c'(x) = 1$ if $x \ge 2c$ and $\psi_c'(x) = 0$ if $x \le 0$; consequently $\psi_c''(x) = 0$ if $x \notin [0, 2c]$, and it suffices to control $\psi_c'$ and $\psi_c''$ in this region. By translation equivariance of the derivative, it then suffices to control the derivatives of $h(x) = x\phi_c(x)$ for $0 < x < c$. We calculate

$$\begin{aligned} h'(x) &= x\phi_c'(x) + \phi_c(x), \\ h''(x) &= x\phi_c''(x) + 2\phi_c(x), \end{aligned} \tag{E.49}$$

and

$$\phi_c'(x) = \frac{f(c - x)f'(x) - f(x)f'(c - x)}{(f(x) + f(c - x))^2}, \tag{E.50}$$

$$\begin{aligned} \phi_c''(x) = & \frac{(f(x) + f(c - x))\left(f(c - x)f''(x) + f(x)f''(c - x) - 2f'(x)f'(c - x)\right)}{(f(x) + f(c - x))^3} \\ & - 2\frac{(f'(x) - f'(c - x))(f(c - x)f'(x) - f(x)f'(c - x))}{(f(x) + f(c - x))^3}. \end{aligned} \tag{E.51}$$

Completely ignoring possible cancellation, we see that it suffices to get a lower bound on $f(x) + f(c - x)$ and upper bounds on $f'$ and $f''$ to bound $|h'|$ and $|h''|$. We calculate

$$f'(x) + f'(c - x) = \frac{1}{x^2} e^{-\frac{1}{x}} \mathbb{1}_{x>0} - \frac{1}{(c - x)^2} e^{-\frac{1}{c-x}} \mathbb{1}_{x<c},$$

and since $f(x) > 0$ if $x > 0$ and $c > 0$, we see that any solution of $f'(x) - f'(c - x) = 0$ must occur for $x \in (0, c)$, which implies as well $c - x \in (0, c)$. Writing $g(x) = x^2 e^{-x}$ and using $c^{-1} < x^{-1} < \infty$ for $x \in (0, c)$, we note from our previous work that $f'(x) - f'(c - x) = 0 \iff$

$g(x^{-1}) = g((c-x)^{-1})$. We calculate $g'(x) = xe^{-x}(2-x)$, so that if $x > 2$ then $g'(x) < 0$, which implies that $g$ is injective on $(2, \infty)$. By assumption, we have $c^{-1} > 2$; consequently there is at most one solution to $f'(x) - f'(c-x) = 0$ in $0 < x < c$, and given that $x = \frac{1}{2}c$ is a solution, there is exactly one solution. We check

$$2f(c/2) < f(0) + f(c) \iff \log 2 < 1/c,$$

where the first RHS is the value of $f(x) + f(c-x)$ at both $x = 0$ and $x = c$, and since $1/c \geq 2$, we conclude that $f(x) + f(c-x) \geq 2f(c/2) > 0$. Next, we use

$$f'(x) = \frac{1}{x^2}e^{-\frac{1}{x}}\mathbb{1}_{x>0},$$

$$f''(x) = \left(\frac{1}{x^4}e^{-\frac{1}{x}} - \frac{2}{x^3}e^{-\frac{1}{x}}\right)\mathbb{1}_{x>0},$$

together with the bound $x^p e^{-x} \leq p^p e^{-p}$ for $p > 0$, which is proved by differentiating $x \mapsto x^p e^{-x}$, equating to zero, and comparing the values of the function at $x = 0$, $x = p$, and $x \to \infty$, to obtain with the triangle inequality

$$|f'(x)| \leq 4/e^2, \qquad |f''(x)| \leq 4^4 e^{-4} + 2 \cdot 3^3 e^{-3}.$$

Combining these bounds with our lower bound on $f(x) + f(c-x)$ and repeatedly applying the triangle inequality and modulus bounds in (E.50) and (E.51), then subsequently in (E.49) (using also $|x| \leq c$), we conclude the claimed bounds on $|\phi_c'|$ and $|\phi_c''|$. □

**Lemma E.32.** *Let $Z, \bar{Z} \in L^2$ be square-integrable random variables. Suppose that $\bar{Z} \leq C$ a.s. and $\|Z - \bar{Z}\|_{L^2} \leq M$. Then*

$$\mathrm{Var}[Z] \leq \mathrm{Var}[\bar{Z}] + CM + M^2.$$

*Proof.* This is a simple consequence of the triangle inequality and the centering inequality for the $L^2$ norm. We have

$$\|Z - \mathbb{E}[Z]\|_{L^2} \leq \|Z - \bar{Z} - \mathbb{E}[Z - \bar{Z}]\|_{L^2} + \|\bar{Z} - \mathbb{E}[\bar{Z}]\|_{L^2},$$

and additionally

$$\|Z - \bar{Z} - \mathbb{E}[Z - \bar{Z}]\|_{L^2} \leq \|Z - \bar{Z}\|_{L^2} \leq M,$$

so that, after squaring, we get

$$\begin{aligned}
\mathrm{Var}[Z] &\leq \mathrm{Var}[\bar{Z}] + M\|\bar{Z} - \mathbb{E}[\bar{Z}]\|_{L^2} + M^2 \\
&\leq \mathrm{Var}[\bar{Z}] + M\|\bar{Z}\|_{L^2} + M^2 \\
&\leq \mathrm{Var}[\bar{Z}] + CM + M^2,
\end{aligned}$$

by centering and the a.s. boundedness assumption. □

**Lemma E.33.** *Let $X, Y$ be square-integrable random variables, and let $d > 0$. Suppose $|X| \leq M_1$ a.s., and suppose $\mathbb{P}[|Y - 1| \geq C\sqrt{d/n}] \leq C'e^{-cd}$ and $\|Y - 1\|_{L^2} \leq M_2$. Then one has with probability at least $1 - C'e^{-cd}$*

$$|XY - \mathbb{E}[XY]| \leq |X - \mathbb{E}[X]| + 2CM_1\sqrt{\frac{d}{n}} + \sqrt{C'}M_1 M_2 e^{-cd/2}.$$

*Proof.* We apply the triangle inequality:

$$\begin{aligned}
|XY - \mathbb{E}[XY]| &\leq |XY - X| + |X - \mathbb{E}[X]| + |\mathbb{E}[X] - \mathbb{E}[XY]| \\
&\leq M_1|Y - 1| + M_1\mathbb{E}[|Y - 1|] + |X - \mathbb{E}[X]|,
\end{aligned}$$

where the second inequality also applies Jensen's inequality. We have

$$\begin{aligned}
\mathbb{E}[|Y - 1|] &= \mathbb{E}\left[\left(\mathbb{1}_{|Y-1|\geq C\sqrt{dn^{-1}}} + \mathbb{1}_{|Y-1|<C\sqrt{d/n}}\right)|Y - 1|\right] \\
&\leq C\sqrt{\frac{d}{n}} + \mathbb{E}\left[\mathbb{1}_{|Y-1|\geq C\sqrt{d/n}}|Y - 1|\right] \\
&\leq C\sqrt{\frac{d}{n}} + \mathbb{E}\left[\mathbb{1}_{|Y-1|\geq C\sqrt{d/n}}\right]^{1/2}\mathbb{E}\left[(Y - 1)^2\right]^{1/2} \\
&\leq C\sqrt{\frac{d}{n}} + \sqrt{C'}e^{-cd/2}M_2,
\end{aligned}$$

where we apply the Schwarz inequality in the third line. Consequently, with probability at least $1 - C'e^{-cd}$, we have

$$|XY - \mathbb{E}[XY]| \leq |X - \mathbb{E}[X]| + 2CM_1\sqrt{\frac{d}{n}} + \sqrt{C'}M_1M_2e^{-cd/2},$$

as claimed.

$\square$

**Lemma E.34.** *For $i = 1, \ldots, n$, let $X_i, Y_i$ be random variables in $L^4$, and let $d > 0$ and $\delta > 0$. Suppose $X_i \geq 0$ for each $i$ and $\sum_{i=1}^{n}\|X_i\|_{L^2} \leq M_3$, and suppose $\mathbb{P}[\forall i \in [n], |Y_i - 1| \geq C\sqrt{d/n}] \leq \delta$ and for each $i$, $\|Y_i - 1\|_{L^4} \leq M_2$. Moreover, suppose that $C\sqrt{d/n} \leq 1$. Then one has with probability at least $1 - \delta$*

$$\left|\sum_{i=1}^{n} X_iY_i - \mathbb{E}[X_iY_i]\right| \leq 2\left|\sum_{i=1}^{n} X_i - \mathbb{E}[X_i]\right| + 2CM_3\sqrt{\frac{d}{n}} + \delta^{1/4}M_2M_3.$$

*Proof.* The proof is a minor elaboration on Lemma E.33. We apply the triangle inequality:

$$\left|\sum_{i=1}^{n} X_iY_i - \mathbb{E}[X_iY_i]\right| \leq \left|\sum_{i=1}^{n} X_iY_i - X_i\right| + \left|\sum_{i=1}^{n} X_i - \mathbb{E}[X_i]\right| + \left|\sum_{i=1}^{n} \mathbb{E}[X_i] - \mathbb{E}[X_iY_i]\right|$$

$$\leq C\left(\sum_{i=1}^{n}|X_i|\right)\sqrt{\frac{d}{n}} + \sum_{i=1}^{n} \mathbb{E}[|X_i||Y_i - 1|] + \left|\sum_{i=1}^{n} X_i - \mathbb{E}[X_i]\right|,$$

where the second line holds with probability at least $1 - \delta$. Another application of the triangle inequality together with nonnegativity of the $X_i$ gives

$$\sum_{i=1}^{n}|X_i| = \left|\sum_{i=1}^{n} X_i\right| \leq \left|\sum_{i=1}^{n} X_i - \mathbb{E}[X_i]\right| + \left|\sum_{i=1}^{n} \mathbb{E}[X_i]\right|$$

$$\leq M_3 + \left|\sum_{i=1}^{n} X_i - \mathbb{E}[X_i]\right|,$$

where the second line applies the Lyapunov inequality. By the Schwarz inequality and the Lyapunov inequality, we have

$$\sum_{i=1}^{n} \mathbb{E}[|X_i||Y_i - 1|] \leq C\sqrt{\frac{d}{n}}\sum_{i=1}^{n} \mathbb{E}[|X_i|] + \sum_{i=1}^{n} \mathbb{E}\left[\mathbb{1}_{|Y_i-1|\geq C\sqrt{d/n}}|X_i||Y_i - 1|\right]$$

$$\leq CM_3\sqrt{\frac{d}{n}} + \delta^{1/4}M_2M_3.$$

Consequently, with probability at least $1 - \delta$, we have

$$\left|\sum_{i=1}^{n} X_iY_i - \mathbb{E}[X_iY_i]\right| \leq 2\left|\sum_{i=1}^{n} X_i - \mathbb{E}[X_i]\right| + 2CM_3\sqrt{\frac{d}{n}} + \delta^{1/4}M_2M_3$$

as claimed, where we use that $C\sqrt{d/n} \leq 1$ here.

$\square$

**Lemma E.35.** *Let $k \in \mathbb{N}$, and let $X_1, \ldots, X_k$ be integrable random variables satisfying $\|X_i - \mathbb{E}[X_i]\|_{L^4} \leq M_i$ for some constants $M_i > 0$. Suppose moreover that with probability at least $1 - \delta_i$, one has $|X_i - \mathbb{E}[X_i]| \leq N_i$ for some constants $N_i > 0$. Then one has*

$$\mathrm{Var}\left[\sum_{i=1}^{k} X_i\right] \leq \sum_{i,j=1}^{k} N_iN_j + \sqrt{\delta_i + \delta_j}M_iM_j.$$

*Proof.* We start from the formula

$$\mathrm{Var}\left[\sum_{i=1}^{k} X_i\right] = \sum_{i=1}^{n} \mathrm{Var}[X_i] + 2\sum_{i<j} \mathrm{cov}[X_i, X_j],$$

where $\mathrm{cov}[X_i, X_j] = \mathbb{E}[X_i X_j] - \mathbb{E}[X_i]\mathbb{E}[X_j] = \mathbb{E}[(X_i - \mathbb{E}[X_i])(X_j - \mathbb{E}[X_j])]$; one establishes this formula by distributing in the definition of the variance. By assumption, there are events $\mathcal{E}_i$ on which $|X_i - \mathbb{E}[X_i]| \leq N_i$ and such that $\mathbb{P}[\mathcal{E}_i] \geq 1 - \delta_i$. Partitioning the expectation, we therefore have

$$\mathrm{Var}[X_i] = \mathbb{E}[(X_i - \mathbb{E}[X_i])^2] \leq N_i^2 + \mathbb{E}[\mathbb{1}_{\mathcal{E}_i^c}(X_i - \mathbb{E}[X_i])^2]$$
$$\leq N_i^2 + \mathbb{E}[\mathbb{1}_{\mathcal{E}_i^c}]^{1/2}\mathbb{E}[(X_i - \mathbb{E}[X_i])^4]^{1/2}$$
$$\leq N_i^2 + \sqrt{\delta_i}M_i^2,$$

where the first line uses nonnegativity of the integrand to discard the indicator after applying the deviations bound, the second line applies the Schwarz inequality, and the third line uses fourth moment control. For the covariance terms, we apply Jensen's inequality to obtain

$$|\mathrm{cov}[X_i, X_j]| = |\mathbb{E}[X_i X_j] - \mathbb{E}[X_i]\mathbb{E}[X_j]| \leq \mathbb{E}[|X_i - \mathbb{E}[X_i]||X_j - \mathbb{E}[X_j]|],$$

so that, again partitioning the outermost expectation and applying our assumptions, we get

$$|\mathrm{cov}[X_i, X_j]| \leq N_i^2 + \mathbb{E}[\mathbb{1}_{\mathcal{E}_i^c \cup \mathcal{E}_j^c}|X_i - \mathbb{E}[X_i]||X_j - \mathbb{E}[X_j]|]$$
$$\leq N_i^2 + \mathbb{E}[\mathbb{1}_{\mathcal{E}_i^c \cup \mathcal{E}_j^c}]^{1/2}\mathbb{E}[(X_i - \mathbb{E}[X_i])^4]^{1/4}\mathbb{E}[(X_j - \mathbb{E}[X_j])^4]^{1/4}$$
$$\leq N_i N_j + \sqrt{\delta_i + \delta_j}M_i M_j,$$

where in the first line we again use nonnegativity of the integrand to discard the indicator after applying the deviations bound, in the second line we apply the Schwarz inequality twice, and in the third line we use a union bound to control the indicator. Since $\delta_i \leq 2\delta_i$, we conclude the claimed expression. $\qquad\square$

**Lemma E.36.** *If $C > 0$ and $p > 0$, the function $g(t) = t^p e^{-Ct^2}$ for $t \geq 0$ satisfies the bound $g(t) \leq (p/(2Ce))^{p/2}$.*

*Proof.* The function $g$ is smooth has derivatives $g'(t) = t^{p-1}e^{-Ct^2}(p - 2Ct^2)$ and $g''(t) = t^{p-2}e^{-Ct^2}(p(p-1) - 2(4p-1)Ct^2 + 4C^2t^4)$. It therefore has at most two critical points, one possibly at $t = 0$ and one at $t = \sqrt{p/(2C)}$, and these points are distinct when $p > 0$ and $C > 0$. We check the sign of $g''$ at the second critical point; since $\sqrt{p/(2C)} > 0$ we need only check the value of $(p(p-1) - 2(4p-1)Ct^2 + 4C^2t^4)$ evaluated at $t = \sqrt{p/(2C)}$, which is $-2p^2 < 0$. Then since $\lim_{t\to\pm\infty} g(t) = 0$ and $g(0) = 0$, we conclude that $g(t) \leq g(\sqrt{p/(2C)})$, which gives the claimed bound. $\qquad\square$

**Lemma E.37.** *Following Lemma E.26, consider the random variables*

$$\Xi_1(s, \boldsymbol{g}_1, \boldsymbol{g}_2) = \sum_{i=1}^{n} \frac{\sigma(g_{1i})^3 \rho(-g_{1i}\cot s)}{\psi(\|\boldsymbol{v}_0\|_2)\psi(\|\boldsymbol{v}_s^i\|_2)\sin^3 s}$$

$$\Xi_2(\nu, \boldsymbol{g}_1, \boldsymbol{g}_2) = \frac{\langle\boldsymbol{v}_0, \boldsymbol{v}_s\rangle\psi'(\|\boldsymbol{v}_s\|_2)\|\boldsymbol{v}_s\|_2}{\psi(\|\boldsymbol{v}_0\|_2)\psi(\|\boldsymbol{v}_s\|_2)^2} - \frac{\langle\boldsymbol{v}_0, \boldsymbol{v}_s\rangle}{\psi(\|\boldsymbol{v}_0\|_2)\psi(\|\boldsymbol{v}_s\|_2)}$$

$$\Xi_3(\nu, \boldsymbol{g}_1, \boldsymbol{g}_2) = -\frac{\langle\boldsymbol{v}_0, \boldsymbol{v}_s\rangle\langle\boldsymbol{v}_s, \dot{\boldsymbol{v}}_s\rangle^2\psi''(\|\boldsymbol{v}_s\|_2)}{\psi(\|\boldsymbol{v}_0\|_2)\psi(\|\boldsymbol{v}_s\|_2)^2\|\boldsymbol{v}_s\|_2^2}$$

$$\Xi_4(\nu, \boldsymbol{g}_1, \boldsymbol{g}_2) = -2\frac{\langle\boldsymbol{v}_0, \dot{\boldsymbol{v}}_s\rangle\langle\boldsymbol{v}_s, \dot{\boldsymbol{v}}_s\rangle\psi'(\|\boldsymbol{v}_s\|_2)}{\psi(\|\boldsymbol{v}_0\|_2)\psi(\|\boldsymbol{v}_s\|_2)^2\|\boldsymbol{v}_s\|_2}$$

$$\Xi_5(\nu, \boldsymbol{g}_1, \boldsymbol{g}_2) = -\frac{\langle\boldsymbol{v}_0, \boldsymbol{v}_s\rangle\|\dot{\boldsymbol{v}}_s\|_2^2\psi'(\|\boldsymbol{v}_s\|_2)}{\psi(\|\boldsymbol{v}_0\|_2)\psi(\|\boldsymbol{v}_s\|_2)^2\|\boldsymbol{v}_s\|_2}$$

$$\Xi_6(\nu, \boldsymbol{g}_1, \boldsymbol{g}_2) = 2\frac{\langle\boldsymbol{v}_0, \boldsymbol{v}_s\rangle\langle\boldsymbol{v}_s, \dot{\boldsymbol{v}}_s\rangle^2\psi'(\|\boldsymbol{v}_s\|_2)}{\psi(\|\boldsymbol{v}_0\|_2)\psi(\|\boldsymbol{v}_s\|_2)^3\|\boldsymbol{v}_s\|_2^2} + \frac{\langle\boldsymbol{v}_0, \boldsymbol{v}_s\rangle\langle\boldsymbol{v}_s, \dot{\boldsymbol{v}}_s\rangle^2\psi'(\|\boldsymbol{v}_s\|_2)}{\psi(\|\boldsymbol{v}_0\|_2)\psi(\|\boldsymbol{v}_s\|_2)^2\|\boldsymbol{v}_s\|_2^3},$$

*where $\Xi_1(\,\cdot\,, \boldsymbol{g}_1, \boldsymbol{g}_2)$ is defined at $\{0, \pi\}$ by continuity (following the proof of Lemma E.27, it is $0$ here). Then for each $i = 1, \ldots, 6$, one has:*

1. *For each $i$, there is a $\mu \otimes \mu$-integrable function $f_i : \mathbb{R}^n \times \mathbb{R}^n \to \mathbb{R}$ such that $|\Xi_i(\,\cdot\,, \boldsymbol{g}_1, \boldsymbol{g}_2) - \mathbb{E}[\Xi_i(\,\cdot\,, \boldsymbol{g}_1, \boldsymbol{g}_2)]| \le f_i(\boldsymbol{g}_1, \boldsymbol{g}_2)$;*

2. *There is an absolute constant $C_i > 0$ such that for every $0 \le \nu \le \pi$, one has $\|f_i\|_{L^4} \le C_i$, so that in particular $\|\Xi_i(\nu, \,\cdot\,, \,\cdot\,) - \mathbb{E}[\Xi_i(\nu, \,\cdot\,, \,\cdot\,)]\|_{L^4} \le C_i$.*

*Proof.* First we reduce to noncentered fourth moment calculations. If $X$ is a random variable with finite fourth moment, we have by Minkowski's inequality

$$\|X - \mathbb{E}[X]\|_{L^4} \le \|X\|_{L^4} + |\mathbb{E}[X]|,$$

so that the triangle inequality for the expectation and the Lyapunov inequality imply

$$\|X - \mathbb{E}[X]\|_{L^4} \le 2\|X\|_{L^4}.$$

We can therefore control the noncentered fourth moments of the random variables $\Xi_i$ and pay only an extra factor of 2 in controlling the centered moments. For the proofs of property 1, we have similarly $|X - \mathbb{E}[X]| \le |X| + \mathbb{E}[|X|]$ from the triangle inequality, so that it again suffices to prove property 1 for the noncentered random variables $|\Xi_i|$.

$\Xi_1$ **control.** If $\nu = 0$ or $\nu = \pi$, the integrand is identically zero; we proceed assuming $0 < \nu < \pi$. Using $\psi \ge \frac{1}{4}$, we have

$$0 \le \Xi_1(\nu, \boldsymbol{g}_1, \boldsymbol{g}_2) \le 16 \sum_{i=1}^{n} \frac{\sigma(g_{1i})^3 \rho(-g_{1i} \cot \nu)}{\sin^3 \nu}.$$

For property 1, by elementary properties of cos we have for $0 \le \nu \le \pi/4$ and $3\pi/4 \le \nu \le \pi$ that $\cos^2 \nu \ge \frac{1}{2}$, so

$$\rho(-g_{1i} \cot \nu) \le \sqrt{\frac{n}{4\pi}} e^{-\frac{n g_{1i}^2}{8 \sin^2 \nu}}.$$

This gives

$$\frac{\sigma(g_{1i})^3 \rho(-g_{1i} \cot \nu)}{\sin^3 \nu} \le \frac{|g_{1i}|^3 \rho(-g_{1i} \cot \nu)}{\sin^3 \nu} = \sqrt{\frac{2}{\pi}} K^{1/2} \left| \frac{g_{1i}}{\sin \nu} \right|^3 e^{-K \left| \frac{g_{1i}}{\sin \nu} \right|^2},$$

where we define $K = n/8$. By Lemma E.36, we have that $g \le g(\sqrt{3/2K}) = CK^{-3/2}$, where $C > 0$ is an absolute constant. We conclude

$$\frac{\sigma(g_{1i})^3 \rho(-g_{1i} \cot \nu)}{\sin^3 \nu} \le C/n, \tag{E.52}$$

provided $\nu$ is not in $[\pi/4, 3\pi/4]$. On the other hand, if $\pi/4 \le \nu \le 3\pi/4$, we have $\sin \nu \ge 1/\sqrt{2}$, so that

$$\frac{\sigma(g_{1i})^3 \rho(-g_{1i} \cot \nu)}{\sin^3 \nu} \le C\sqrt{n} \sigma(g_{1i})^3, \tag{E.53}$$

where $C > 0$ is an absolute constant. Since these $\nu$ constraints cover $[0, \pi]$, we have for all $\nu$ and all $\boldsymbol{g}_1$ (by the triangle inequality)

$$|\Xi_1(\nu, \boldsymbol{g}_1, \boldsymbol{g}_2)| \le C + C' n^{3/2} \sigma(g_{1i})^3,$$

where $C, C' > 0$ are absolute constants, and by Lemma G.11, we have

$$\mathbb{E}[C + C' n^{3/2} \sigma(g_{1i})^3] = C + C'',$$

where $C'' > 0$ is an absolute constant. This proves property 1 with $f_1(\boldsymbol{g}_1, \boldsymbol{g}_2) = C + C' n^{3/2} \sigma(g_{1i})^3$, with different absolute constants, and property 2 follows from Lemma G.11 after applying the Minkowski inequality and calculating the integral, which has the necessary cancellation of the $n^{3/2}$ factor.

$\Xi_2$ **control.** By Lemma E.31, we have $|\psi'| \le C$ for an absolute constant $C > 0$ and $x/\psi(x) \le 2$. Cauchy-Schwarz then implies

$$\left| \frac{\langle \boldsymbol{v}_0, \boldsymbol{v}_s \rangle \psi'(\|\boldsymbol{v}_s\|_2) \|\boldsymbol{v}_s\|_2}{\psi(\|\boldsymbol{v}_0\|_2) \psi(\|\boldsymbol{v}_s\|_2)^2} \right| \le 8C.$$

In an exactly analogous manner, we have

$$\left| \frac{\langle \boldsymbol{v}_0, \boldsymbol{v}_s \rangle}{\psi(\|\boldsymbol{v}_0\|_2) \psi(\|\boldsymbol{v}_s\|_2)} \right| \le 4.$$

Both bounds satisfy the requirements of property 1, with $f_2(\boldsymbol{g}_1, \boldsymbol{g}_2) = 16C + 8$. The triangle inequality and Minkowski's inequality then implies $\|\Xi_1(\nu, \cdot, \cdot)\|_{L^4} \le C'$.

$\Xi_3$ **control.** By Lemma E.31, we have $|\psi''| \le C$ for an absolute constant $C > 0$, $\psi \ge \frac{1}{4}$, and $x/\psi(x) \le 2$. Cauchy-Schwarz then implies

$$\left| \frac{\langle \boldsymbol{v}_0, \boldsymbol{v}_s \rangle \langle \boldsymbol{v}_s, \dot{\boldsymbol{v}}_s \rangle^2 \psi''(\|\boldsymbol{v}_s\|_2)}{\psi(\|\boldsymbol{v}_0\|_2) \psi(\|\boldsymbol{v}_s\|_2)^2 \|\boldsymbol{v}_s\|_2^2} \right| \le 16C\|\dot{\boldsymbol{v}}_s\|_2^2,$$

and the triangle inequality gives $\|\dot{\boldsymbol{v}}_s\|_2^2 \le \|\boldsymbol{g}_1\|_2^2 + \|\boldsymbol{g}_2\|_2^2 + 2\|\boldsymbol{g}_1\|_2 \|\boldsymbol{g}_2\|_2$, whose expectation is bounded by 4, by the Schwarz inequality and Lemma G.11. We can therefore take $f_3(\boldsymbol{g}_1, \boldsymbol{g}_2) = C + C'(\|\boldsymbol{g}_1\|_2 + \|\boldsymbol{g}_2\|_2)^2$, and we have

$$\left\| (\|\boldsymbol{g}_1\|_2 + \|\boldsymbol{g}_2\|_2)^2 \right\|_{L^4} = \left\| \|\boldsymbol{g}_1\|_2 + \|\boldsymbol{g}_2\|_2 \right\|_{L^8}^2 \le \left( \|\|\boldsymbol{g}_1\|_2\|_{L^8} + \|\|\boldsymbol{g}_2\|_2\|_{L^8} \right)^2 \le C,$$

where $C > 0$ is a (new) absolute constant, by the Minkowski inequality and lemmas Lemmas G.10 and G.11. This establishes property 2.

$\Xi_4$ **control.** By Lemma E.31, we have $|\psi'| \le C$ for an absolute constant $C > 0$, $\psi \ge \frac{1}{4}$, and $x/\psi(x) \le 2$; Cauchy-Schwarz then implies

$$\left| 2 \frac{\langle \boldsymbol{v}_0, \dot{\boldsymbol{v}}_s \rangle \langle \boldsymbol{v}_s, \dot{\boldsymbol{v}}_s \rangle \psi'(\|\boldsymbol{v}_s\|_2)}{\psi(\|\boldsymbol{v}_0\|_2) \psi(\|\boldsymbol{v}_s\|_2)^2 \|\boldsymbol{v}_s\|_2} \right| \le 64C\|\dot{\boldsymbol{v}}_s\|_2^2.$$

Following the argument for $\Xi_3$ exactly, we conclude property 1 and 2 from this bound with a suitable modification of the constant.

$\Xi_5$ **control.** We have

$$\left| \frac{\langle \boldsymbol{v}_0, \boldsymbol{v}_s \rangle \|\dot{\boldsymbol{v}}_s\|_2^2 \psi'(\|\boldsymbol{v}_s\|_2)}{\psi(\|\boldsymbol{v}_0\|_2) \psi(\|\boldsymbol{v}_s\|_2)^2 \|\boldsymbol{v}_s\|_2} \right| \le 32C\|\dot{\boldsymbol{v}}_s\|_2^2,$$

following exactly the setup and instantiations in the argument for $\Xi_4$. Following the argument for $\Xi_3$ exactly, we conclude property 1 and 2 from this bound with a suitable modification of the constant.

$\Xi_6$ **control.** The triangle inequality gives

$$|\Xi_6(s, \boldsymbol{g}_1, \boldsymbol{g}_2)| \le 2 \left| \frac{\langle \boldsymbol{v}_0, \boldsymbol{v}_s \rangle \langle \boldsymbol{v}_s, \dot{\boldsymbol{v}}_s \rangle^2 \psi'(\|\boldsymbol{v}_s\|_2)}{\psi(\|\boldsymbol{v}_0\|_2) \psi(\|\boldsymbol{v}_s\|_2)^3 \|\boldsymbol{v}_s\|_2^2} \right| + \left| \frac{\langle \boldsymbol{v}_0, \boldsymbol{v}_s \rangle \langle \boldsymbol{v}_s, \dot{\boldsymbol{v}}_s \rangle^2 \psi'(\|\boldsymbol{v}_s\|_2)}{\psi(\|\boldsymbol{v}_0\|_2) \psi(\|\boldsymbol{v}_s\|_2)^2 \|\boldsymbol{v}_s\|_2^3} \right|,$$

and following the setup of $\Xi_4$ and $\Xi_5$ control gives $|\Xi_6(\nu, \boldsymbol{g}_1, \boldsymbol{g}_2)| \le 128C\|\dot{\boldsymbol{v}}_\nu\|_2^2 + 32C\|\dot{\boldsymbol{v}}_\nu\|_2^2$. Following the argument for $\Xi_3$ exactly, we conclude property 1 and 2 from this bound with a suitable modification of the constant. $\square$

**Lemma E.38.** *In the notation of Lemma E.13, there are absolute constants $c, c', C > 0$ and an absolute constant $K > 0$ such that if $n \ge K$, there is an event with probability at least $1 - 2e^{-cn}$ on which one has*

$$\left| \frac{\|\boldsymbol{v}_0\|_2^2}{\psi(\|\boldsymbol{v}_0\|_2)^2} - \mathbb{E}\left[ \frac{\|\boldsymbol{v}_0\|_2^2}{\psi(\|\boldsymbol{v}_0\|_2)^2} \right] \right| \le Ce^{-c'n}.$$

*Proof.* There is no $\nu$ dependence in this term, so we need only prove a single bound. Following the proof of the measure bound in Lemma E.16, but using only the pointwise concentration result, we assert that if $n \ge C$ an absolute constant there is an event $\mathcal{E}$ on which $0.5 \le \|\boldsymbol{v}_0\|_2 \le 2$ with probability at least $1 - 2e^{-cn}$ with $c > 0$ an absolute constant. This implies that if $\boldsymbol{g}_1 \in \mathcal{E}$ we have

$$\frac{\|\boldsymbol{v}_0\|_2^2}{\psi(\|\boldsymbol{v}_0\|_2)^2} = 1,$$

which we can use together with nonnegativity of the integrand to calculate

$$\mathbb{E}\left[\frac{\|\boldsymbol{v}_0\|_2^2}{\psi(\|\boldsymbol{v}_0\|_2)^2}\right] = \mathbb{E}[\mathbb{1}_{\mathcal{E}}] + \mathbb{E}\left[\mathbb{1}_{\mathcal{E}^c}\left(\frac{\|\boldsymbol{v}_0\|_2}{\psi(\|\boldsymbol{v}_0\|_2)}\right)^2\right]$$

$$\geq \mathbb{E}[\mathbb{1}_{\mathcal{E}}] \geq 1 - 2e^{-cn},$$

whence

$$\frac{\|\boldsymbol{v}_0\|_2^2}{\psi(\|\boldsymbol{v}_0\|_2)^2} - \mathbb{E}\left[\frac{\|\boldsymbol{v}_0\|_2^2}{\psi(\|\boldsymbol{v}_0\|_2)^2}\right] \leq 2e^{-cn}$$

whenever $\boldsymbol{g}_1 \in \mathcal{E}$. Similarly, we calculate

$$\mathbb{E}\left[\frac{\|\boldsymbol{v}_0\|_2^2}{\psi(\|\boldsymbol{v}_0\|_2)^2}\right] = \mathbb{E}[\mathbb{1}_{\mathcal{E}}] + \mathbb{E}\left[\mathbb{1}_{\mathcal{E}^c}\left(\frac{\|\boldsymbol{v}_0\|_2}{\psi(\|\boldsymbol{v}_0\|_2)}\right)^2\right]$$

$$\leq 1 + \mathbb{E}[\mathbb{1}_{\mathcal{E}^c}]^{1/2}\mathbb{E}\left[\left(\frac{\|\boldsymbol{v}_0\|_2}{\psi(\|\boldsymbol{v}_0\|_2)}\right)^4\right]^{1/2}$$

$$\leq 1 + 16Ce^{-cn},$$

applying the Schwarz inequality, property 2 in Lemma E.31, and the measure bound on $\mathcal{E}$, with $c', C' > 0$ absolute constants, whence

$$\mathbb{E}\left[\frac{\|\boldsymbol{v}_0\|_2^2}{\psi(\|\boldsymbol{v}_0\|_2)^2}\right] - \frac{\|\boldsymbol{v}_0\|_2^2}{\psi(\|\boldsymbol{v}_0\|_2)^2} \leq 16C'e^{-c'n}$$

whenever $\boldsymbol{g}_1 \in \mathcal{E}$. Worst-casing constants, we conclude

$$\left|\frac{\|\boldsymbol{v}_0\|_2^2}{\psi(\|\boldsymbol{v}_0\|_2)^2} - \mathbb{E}\left[\frac{\|\boldsymbol{v}_0\|_2^2}{\psi(\|\boldsymbol{v}_0\|_2)^2}\right]\right| \leq Ce^{-cn}$$

when $\boldsymbol{g}_1 \in \mathcal{E}$, which is sufficient for our purposes.

$\square$

**Lemma E.39.** *In the notation of Lemma E.13, if $d \geq 1$, there are absolute constants $c, c', c'', c''', C, C', C'', C''', C''''  > 0$ and absolute constants $K, K' > 0$ such that if $n \geq Kd^4\log^4 n$ and $d \geq K'$, there is an event with probability at least $1 - C'''n^{-c''\bar{d}/2} - C''''ne^{-c'''n}$ on which one has*

$$|\Xi_1(\nu, \boldsymbol{g}_1, \boldsymbol{g}_2) - \mathbb{E}[\Xi_1(\nu, \boldsymbol{g}_1, \boldsymbol{g}_2)]| \leq C\sqrt{\frac{\bar{d}\log n}{n}} + C'n^{-c\bar{d}} + C''ne^{-c'n}.$$

*Proof.* If $\nu \in \{0, \pi\}$, then $\Xi_1(\nu, \boldsymbol{g}_1, \boldsymbol{g}_2) = 0$ for every $(\boldsymbol{g}_1, \boldsymbol{g}_2)$; we therefore assume $0 < \nu < \pi$ below.

We will apply Lemma E.34 to begin, with the instantiations

$$X_i = \frac{\sigma(g_{1i})^3\rho(-g_{1i}\cot\nu)}{\sin^3\nu}, \qquad Y_i = \frac{1}{\psi(\|\boldsymbol{v}_0\|_2)\psi(\|\boldsymbol{v}_\nu^i\|_2)},$$

since then $\Xi_1(\nu, \boldsymbol{g}_1, \boldsymbol{g}_2) = \sum_i X_iY_i$. We have $X_i \geq 0$; writing $k^2 = 2/n$, we calculate

$$\mathbb{E}[X_i] = \frac{1}{\sqrt{8\pi k^2}}\frac{1}{\sqrt{2\pi k^2}}\int_{\mathbb{R}}\frac{g^3}{\sin^3\nu}\exp\left(-\frac{1}{2k^2}\frac{g^2}{\sin^2\nu}\right)\mathrm{d}g$$

$$= \frac{2}{\pi n}\sin\nu \qquad\qquad\qquad (\text{E.54})$$

where the second line uses the change of variables $g \mapsto g \sin \nu$ and Lemma G.11. Additionally, we have

$$
\begin{aligned}
\mathbb{E}[X_i^2] &= \frac{k^4}{4\pi} \frac{1}{\sqrt{2\pi}} \int_{\mathbb{R}} \frac{g^6}{\sin^6 \nu} \exp\left(-\frac{1}{2} g^2 (1 + 2\cot^2 \nu)\right) \mathrm{d}g \\
&= \frac{k^4 \sin \nu}{4\pi} \frac{1}{\sqrt{2\pi}} \int_{\mathbb{R}} g^6 \exp\left(-\frac{1}{2} g^2 (1 + \cos^2 \nu)\right) \mathrm{d}g \\
&= \frac{k^4 \sin \nu}{4\pi (1 + \cos^2 \nu)^{7/2}} \frac{1}{\sqrt{2\pi}} \int_{\mathbb{R}} g^6 e^{-g^2/2} \mathrm{d}g \\
&= \frac{15 \sin \nu}{\pi n^2 (1 + \cos^2 \nu)^{7/2}},
\end{aligned}
\tag{E.55}
$$

where in the second line we change variables $g \mapsto g \sin \nu$, in the third line we change variables $g \mapsto g/\sqrt{1 + \cos^2 \nu}$, and in the fourth line we use Lemma G.11. We can calculate the derivative of the map $g(\nu) = (1 + \cos^2 \nu)^{-7/2} \sin \nu$ as $g'(\nu) = \cos(\nu)(1 + \cos^2 \nu)^{-7}[(1 + \cos^2 \nu)^{7/2} + 7\sin^2(\nu)(1 + \cos^2 \nu)^{5/2}]$, which evidently has the same sign as $\cos(\nu)$; so $g$ is strictly increasing below $\pi/2$ and strictly decreasing above it, and is therefore maximized at $g(\pi/2)$. We conclude the bound

$$
\mathbb{E}[X_i^2] \leq \frac{15}{\pi n^2},
\tag{E.56}
$$

which shows that $\sum_i \|X_i\|_{L^2} = O(1)$. Next, we have $Y_i \leq 16$ by Lemma E.31, so by the Minkowski inequality $\|Y_i - 1\|_{L^4} \leq 17$ for each $i$, and it remains to control deviations. We consider the event $\mathcal{E} = \mathcal{E}_{0.5, 1}$ in the notation of Lemma E.16, which has probability at least $1 - Cne^{-cn}$ and on which we have $\frac{1}{2} \leq \|v_\nu^i\|_2 \leq 2$ for all $i \in [n]$ and in particular $\frac{1}{2} \leq \|v_0\|_2$, and thus by Lemma E.31

$$
Y_i = \frac{1}{\|v_0\|_2 \|v_\nu^i\|_2}
$$

for all $i \in [n]$. By Taylor expansion with Lagrange remainder of the smooth function $x \mapsto x^{-1}$ on the domain $x > 0$ about the point 1, we have

$$
\frac{1}{x} = 1 - (x - 1) + \frac{1}{\xi^3}(x - 1)^2,
$$

where $\xi$ lies between 1 and $x$. If $(g_1, g_2) \in \mathcal{E}$, then for all $i$ $\|v_0\|_2^3 \|v_\nu^i\|_2^3 \geq (1/64)$, and we can therefore assert

$$
\left(\|v_0\|_2 \|v_\nu^i\|_2 - 1\right) - 64\left(\|v_0\|_2 \|v_\nu^i\|_2 - 1\right)^2 \leq 1 - Y_i \leq \left(\|v_0\|_2 \|v_\nu^i\|_2 - 1\right).
\tag{E.57}
$$

By Gauss-Lipschitz concentration, we have $\mathbb{P}[|\|v_0\|_2 - \mathbb{E}[\|v_0\|_2]| \geq t] \leq 2e^{-cnt^2}$ and $\mathbb{P}[|\|v_0\|_2 - \mathbb{E}[\|v_\nu^i\|_2]| \geq t] \leq 2e^{-cnt^2}$. Lemma E.19 implies that $1 - 2/(n-1) \leq \mathbb{E}[\|v_\nu^i\|_2] \leq 1$ and $1 - 2/n \leq \mathbb{E}[\|v_0\|_2] \leq 1$, so we can conclude when $n \geq d$ and when $n$ is larger than a constant that

$$
|\|v_0\|_2 - 1| \leq C\sqrt{\frac{d}{n}}; \qquad \forall i \in [n], \, |\|v_\nu^i\|_2 - 1| \leq C\sqrt{\frac{d}{n}}
$$

with probability at least $1 - C'ne^{-d}$, by a union bound. Using then the fact that $\|v_\nu^i\|_2 \leq 2$ for all $i$ on the event $\mathcal{E}$ together with the previous estimates and (E.57), we obtain with probability at least $1 - C''ne^{-cn} - C'''ne^{-d}$ (via a union bound with the measure of $\mathcal{E}$) that for all $i$,

$$
C\sqrt{\frac{d}{n}} - C'\frac{d}{n} \leq 1 - Y_i \leq C\sqrt{\frac{d}{n}}.
$$

As long as $n \geq d$, we conclude that with the same probability, for all $i$ we have $|Y_i - 1| \leq C\sqrt{d/n}$. We can therefore apply Lemma E.34 to get that with probability at least $1 - C''ne^{-cn} - C'''ne^{-d}$ we have

$$
\begin{aligned}
|\Xi_1(\nu, g_1, g_2) - \mathbb{E}[\Xi_1(\nu, g_1, g_2)]| \leq \, &2\left|\sum_{i=1}^{n} \frac{\sigma(g_{1i})^3 \rho(-g_{1i} \cot \nu)}{\sin^3 \nu} - \mathbb{E}\left[\frac{\sigma(g_{1i})^3 \rho(-g_{1i} \cot \nu)}{\sin^3 \nu}\right]\right| \\
&+ C\sqrt{\frac{d}{n}} + (C'')^{1/4} ne^{-cn/4} + (C''')^{1/4} ne^{-d/4},
\end{aligned}
\tag{E.58}
$$

where we also used the triangle inequality for the $\ell^4$ norm to simplify the fourth root term, together with $n \geq 1$. For $\nu \in [0, \pi]$, we define $f_\nu : \mathbb{R} \to \mathbb{R}$ by

$$f_\nu(g) = \frac{\sigma(g)^3}{\sqrt{2\pi k^2 \sin^3 \nu}} \exp\left(-\frac{1}{2k^2} g^2 \cot^2 \nu\right),$$

so that the task that remains is to control $|\sum_i f_\nu(g_{1i}) - \mathbb{E}[f_\nu(g_{1i})]|$. We start by applying Lemma E.36 to obtain an estimate

$$f_\nu(g) \leq \frac{C}{n|\cos \nu|^3},$$

where $C > 0$ is an absolute constant. When $0 \leq \nu \leq \pi/4$ or $3\pi/4 \leq \nu \leq \pi$, we have therefore $f_\nu(g) \leq C/n$. Meanwhile, if $\pi/4 \leq \nu \leq 3\pi/4$, we have $f_\nu(g) \leq C'\sqrt{n}\sigma(g)^3$, so we can conclude $f_\nu(g) \leq C/n + C'\sqrt{n}\sigma(g)^3$ for all $\nu$, which shows that $f_\nu(g)$ is not much larger than $C'\sqrt{n}\sigma(g)^3$.

Next, let $\bar{g} \sim \mathcal{N}(0, 1)$, so that $g \stackrel{d}{=} k\bar{g}$; we have for any $t \geq 0$

$$\mathbb{P}\big[C'\sqrt{n}\sigma(g)^3 \geq t\big] = \mathbb{P}\Big[\sigma(\bar{g}) \geq C''(nt)^{1/3}\Big] \leq \exp\left(-\tfrac{1}{2}(C'')^2(nt)^{2/3}\right),$$

where we use the classical estimate $\mathbb{P}[\bar{g} \geq t] \leq e^{-t^2/2}$, valid for $t \geq 1$, and accordingly require $t \geq (C'')^{-3}n^{-1}$. In particular, there is an absolute constant $C'' > 0$ such that we have

$$\mathbb{P}\left[C'\sqrt{n}\sigma(g)^3 \geq \frac{C''}{\sqrt{nd}}\right] = \mathbb{P}\left[\sigma(\bar{g}) \geq \left(\frac{n}{d}\right)^{1/6}\right] \leq \exp\left(-\frac{1}{2}\left(\frac{n}{d}\right)^{1/3}\right) \leq e^{-d},$$

where the last inequality holds in particular when $n \geq 8d^4$ (and this condition implies what is necessary for the second to last to hold when $d \geq 1$). Returning to our bound on $f_\nu$, we note that when $n \geq (C/C'')^2 d$, we have that

$$f_\nu(g) - \frac{2C''}{\sqrt{nd}} \leq \frac{C}{n} + C'\sqrt{n}\sigma(g)^3 - \frac{2C''}{\sqrt{nd}} \leq C'\sqrt{n}\sigma(g)^3 - \frac{C''}{\sqrt{nd}},$$

from which we conclude that when our previous hypotheses on $n$ are in force

$$\mathbb{P}\left[f_\nu(g) \geq \frac{2C''}{\sqrt{nd}}\right] \leq e^{-d}. \tag{E.59}$$

We are going to use this result to control $|\sum_i f_\nu(g_{1i}) - \mathbb{E}[f_\nu(g_{1i})]|$ using a truncation approach. Define $M = 2C''/\sqrt{nd}$, where $C'' > 0$ is the absolute constant in (E.59). We write using the triangle inequality

$$\left|\sum_{i=1}^n f_\nu(g_{1i}) - \mathbb{E}[f_\nu(g_{1i})]\right| \leq \left|\sum_{i=1}^n f_\nu(g_{1i}) - f_\nu(g_{1i})\mathbb{1}_{f_\nu(g_{1i}) \leq M}\right|$$

$$+ \left|\sum_{i=1}^n f_\nu(g_{1i})\mathbb{1}_{f_\nu(g_{1i}) \leq M} - \mathbb{E}\big[f_\nu(g_{1i})\mathbb{1}_{f_\nu(g_{1i}) \leq M}\big]\right|$$

$$+ \left|\sum_{i=1}^n \mathbb{E}\big[f_\nu(g_{1i})\mathbb{1}_{f_\nu(g_{1i}) \leq M}\big] - \mathbb{E}[f_\nu(g_{1i})]\right|.$$

By (E.59) and a union bound, we have with probability at least $1 - ne^{-d}$

$$\left|\sum_{i=1}^n f_\nu(g_{1i}) - f_\nu(g_{1i})\mathbb{1}_{f_\nu(g_{1i}) \leq M}\right| = 0.$$

Moreover, we calculate

$$\left|\sum_{i=1}^n \mathbb{E}\big[f_\nu(g_{1i})\mathbb{1}_{f_\nu(g_{1i}) \leq M}\big] - \mathbb{E}[f_\nu(g_{1i})]\right| \leq \sum_{i=1}^n \mathbb{E}\big[f_\nu(g_{1i})\mathbb{1}_{f_\nu(g_{1i}) > M}\big]$$

$$\leq \sum_{i=1}^n \mathbb{P}[f_\nu(g_{1i}) > M]^{1/2} \|f_\nu(g_{1i})\|_{L^2}$$

$$\leq C e^{-d/2}$$

for an absolute constant $C > 0$, using in the second line the Schwarz inequality, and in the third line (E.56) and (E.59). The second term can be controlled with Lemma G.3, together with the observation that

$$\sum_{i=1}^{n} \mathbb{E}\Big[\big(\mathbb{1}_{f_\nu(g_{1i}) \leq M} f_\nu(g_{1i}) - \mathbb{E}\big[\mathbb{1}_{f_\nu(g_{1i}) \leq M} f_\nu(g_{1i})\big]\big)^2\Big]$$

$$= \sum_{i=1}^{n} \mathbb{E}\big[\mathbb{1}_{f_\nu(g_{1i}) \leq M} f_\nu(g_{1i})^2\big] - \mathbb{E}\big[\mathbb{1}_{f_\nu(g_{1i}) \leq M} f_\nu(g_{1i})\big]^2$$

$$\leq \sum_{i=1}^{n} \mathbb{E}\big[f_\nu(g_{1i})^2\big]$$

$$\leq C/n,$$

where the last inequality is due to (E.56). Lemma G.3 thus gives for any $t \geq 0$

$$\mathbb{P}\left[\left|\sum_{i=1}^{n} f_\nu(g_{1i})\mathbb{1}_{f_\nu(g_{1i}) \leq M} - \mathbb{E}\big[f_\nu(g_{1i})\mathbb{1}_{f_\nu(g_{1i}) \leq M}\big]\right| \geq t\right] \leq 2\exp\left(-\frac{t^2/2}{Cn^{-1} + Mt/3}\right).$$

It follows that there is an absolute constant $C' > 0$ such that

$$\mathbb{P}\left[\left|\sum_{i=1}^{n} f_\nu(g_{1i})\mathbb{1}_{f_\nu(g_{1i}) \leq M} - \mathbb{E}\big[f_\nu(g_{1i})\mathbb{1}_{f_\nu(g_{1i}) \leq M}\big]\right| \geq C'\sqrt{\frac{d}{n}}\right] \leq 2e^{-d},$$

and therefore with probability at least $1 - 2ne^{-d}$ (by a union bound) we have

$$\left|\sum_{i=1}^{n} f_\nu(g_{1i}) - \mathbb{E}[f_\nu(g_{1i})]\right| \leq C'\sqrt{\frac{d}{n}} + C''e^{-d/2}.$$

Combining with (E.58) using a union bound and worst-casing constants in the exponent, we conclude that with probability at least $1 - C'''ne^{-c''d} - C''''ne^{-c'''n}$, we have

$$|\Xi_1(\nu, \boldsymbol{g}_1, \boldsymbol{g}_2) - \mathbb{E}[\Xi_1(\nu, \boldsymbol{g}_1, \boldsymbol{g}_2)]| \leq C\sqrt{\frac{d}{n}} + C'e^{-cd} + C''ne^{-c'n}.$$

Aggregating our hypotheses on $n$, there are absolute constants $C_1, C_2, C_3 > 0$ such that we have to satisfy $n \geq \max\{Cd, C'd^4, C''\}$. Moreover, to be able to assert $ne^{-c''d} \leq e^{-c''d/2}$, we have to satisfy $d \geq 2/c'' \log n$. Introducing an auxiliary $\bar{d} > 0$ and setting $d = \bar{d}\log n$, we have to satisfy $n \geq \max\{C\bar{d}\log n, C'\bar{d}^4 \log^4 n, C''\}$ and $\bar{d} \geq 2/c''$. Choosing $n$ in this way, we can finally conclude that with probability at least $1 - C'''n^{-c''\bar{d}/2} - C''''ne^{-c'''n}$, we have

$$|\Xi_1(\nu, \boldsymbol{g}_1, \boldsymbol{g}_2) - \mathbb{E}[\Xi_1(\nu, \boldsymbol{g}_1, \boldsymbol{g}_2)]| \leq C\sqrt{\frac{\bar{d}\log n}{n}} + C'n^{-c\bar{d}} + C''ne^{-c'n},$$

which is the desired type of bound.

$\qquad\square$

**Lemma E.40.** *In the notation of Lemma E.13, there are absolute constants $c, C, C', C'' > 0$ such that for any $\delta \geq 3/2$, we have*

$$\mathbb{P}\left[\left|\mathbb{E}_{\boldsymbol{g}_2}[\Xi_1(\nu, \boldsymbol{g}_1, \boldsymbol{g}_2)] - \mathbb{E}_{\boldsymbol{g}_1, \boldsymbol{g}_2}[\Xi_1(\nu, \boldsymbol{g}_1, \boldsymbol{g}_2)]\right| \text{ is } C + C'n^{1+\delta}\text{-Lipschitz}\right] \geq 1 - 2e^{-cn} - C''n^{-\delta}.$$

*Proof.* Write $f(\nu, \boldsymbol{g}_1) = \mathbb{E}_{\boldsymbol{g}_2}[\Xi_1(\nu, \boldsymbol{g}_1, \boldsymbol{g}_2)]$; it will suffice to differentiate $f$ and $\mathbb{E}[f]$ with respect to $\nu$, bound the derivatives on an event of high probability, and apply the triangle inequality to obtain a high-probability Lipschitz estimate for $|\mathbb{E}_{\boldsymbol{g}_2}[\Xi_1(\nu, \boldsymbol{g}_1, \boldsymbol{g}_2)] - \mathbb{E}_{\boldsymbol{g}_1, \boldsymbol{g}_2}[\Xi_1(\nu, \boldsymbol{g}_1, \boldsymbol{g}_2)]|$.

Define $k = \sqrt{2/n}$. For fixed $(\boldsymbol{g}_1, \boldsymbol{g}_2)$, the function

$$q(\nu, \boldsymbol{g}_1, \boldsymbol{g}_2) = \sum_{i=1}^{n} \frac{\sigma(g_{1i})^3 \rho(-g_{1i}\cot\nu)}{\psi(\|\boldsymbol{v}_0\|_2)\psi(\|\boldsymbol{v}_\nu^i\|_2)\sin^3\nu}$$

is differentiable at all but at most $n$ points of $(0, \nu)$, using Lemma E.31 to see that the only obstruction to differentiability is the function $\sigma$ in the term $\|\boldsymbol{v}_\nu^i\|_2$; and there has derivative

$$q'(\nu, \boldsymbol{g}_1, \boldsymbol{g}_2) = \sum_{i=1}^n \frac{\sigma(g_{1i})^3}{\sqrt{2\pi k^2}\psi(\|\boldsymbol{v}_0\|_2)} \left( \begin{array}{c} \frac{g_{1i}^2 \cos \nu}{k^2 \psi(\|\boldsymbol{v}_\nu^i\|_2)\sin^6 \nu} \\ -\frac{\psi'(\|\boldsymbol{v}_\nu^i\|_2)\langle \boldsymbol{v}_\nu^i, \dot{\boldsymbol{v}}_\nu^i\rangle}{\psi(\|\boldsymbol{v}_\nu^i\|_2)^2 \|\boldsymbol{v}_\nu^i\|_2 \sin^3 \nu} \\ -3\frac{\cos \nu}{\psi(\|\boldsymbol{v}_\nu^i\|_2)\sin^4 \nu} \end{array} \right) \exp\left( -\frac{1}{2k^2}g_{1i}^2 \cot^2 \nu \right).$$

The triangle inequality and Lemma E.31 yield

$$|q'(\nu, \boldsymbol{g}_1, \boldsymbol{g}_2)| \leq \frac{4}{\sqrt{2\pi k^2}} \sum_{i=1}^n |g_{1i}|^3 \left( \frac{4g_{1i}^2}{k^2 \sin^6 \nu} + \frac{16C\|\dot{\boldsymbol{v}}_\nu^i\|_2}{\sin^3 \nu} + \frac{12}{\sin^4 \nu} \right) \exp\left( -\frac{1}{2k^2}g_{1i}^2 \cot^2 \nu \right)$$
(E.60)

for $C > 0$ an absolute constant. We have $\|\dot{\boldsymbol{v}}_\nu^i\|_2 \leq \|\boldsymbol{g}_1\|_2 + \|\boldsymbol{g}_2\|_2$ by the triangle inequality, so to obtain a $(\nu, \boldsymbol{g}_2)$-integrable upper bound it suffices to remove the $\nu$ dependence from the previous estimate. We argue as follows: if $0 \leq \nu \leq \pi/4$ or $3\pi/4 \leq \nu \leq \pi$, we have $\cos^2 \nu \geq \frac{1}{2}$, and so for any $p \geq 3$

$$\frac{\exp\left( -\frac{1}{2k^2}g_{1i}^2 \cot^2 \nu \right)}{\sin^p \nu} \leq \exp\left( \frac{g_{1i}^2}{4k^2}\frac{1}{\sin^2 t} \right) \sin^{-p} \nu.$$
(E.61)

By Lemma E.36, where we put $C = g_{1i}^2/4k^2$ and therefore have to require that $g_{1i} \neq 0$ for all $i \in [n]$ (a set of measure zero in $\mathbb{R}^n$), this yields

$$|q'(\nu, \boldsymbol{g}_1, \boldsymbol{g}_2)| \leq C\|\boldsymbol{g}_2\|_2,$$
(E.62)

where $C > 0$ is a constant depending only on $n$ and $\boldsymbol{g}_1$. In cases where $g_{1i} = 0$ for some $i$, we note that the bound (E.60) is then equal to zero, which also satisfies the estimate (E.62). On the other hand, when $\pi/4 \leq \nu \leq 3\pi/4$, then $\sin t \geq 2^{-1/2}$, and we can assert for any $p \geq 3$

$$\frac{\exp\left( -\frac{1}{2k^2}g_{1i}^2 \cot^2 \nu \right)}{\sin^p \nu} \leq 2^{p/2}.$$

By the triangle inequality, this too implies

$$|q'(\nu, \boldsymbol{g}_1, \boldsymbol{g}_2)| \leq C'\|\boldsymbol{g}_2\|_2,$$

where $C' > 0$ is a constant depending only on $n$ and $\boldsymbol{g}_1$. Invoking then Lemma G.9, we conclude that $q'$ is absolutely integrable over $[0, \pi] \times \mathbb{R}^n$, so that an application of Fubini's theorem and (Cohn, 2013, Theorem 6.3.11) gives the Taylor expansion $f(\nu, \boldsymbol{g}_1) = f(0, \boldsymbol{g}_1) + \int_0^\nu \mathbb{E}_{\boldsymbol{g}_2}[q'(t, \boldsymbol{g}_1, \boldsymbol{g}_2)]\,\mathrm{d}t$. Next, we show also that $q'$ is absolutely integrable over $[0, \pi] \times \mathbb{R}^n \times \mathbb{R}^n$, which implies that $\mathbb{E}[f(\nu, \boldsymbol{g}_1)] = \mathbb{E}[f(0, \boldsymbol{g}_1)] + \int_0^\nu \mathbb{E}[q'(t, \boldsymbol{g}_1, \boldsymbol{g}_2)]\,\mathrm{d}t$ as well. Starting from (E.60), we have

$$\mathbb{E}[|q'(\nu, \boldsymbol{g}_1, \boldsymbol{g}_2)|]$$
$$\leq \mathbb{E}\left[ \frac{4}{\sqrt{2\pi k^2}} \sum_{i=1}^n |g_{1i}|^3 \left( \frac{4g_{1i}^2}{k^2 \sin^6 \nu} + \frac{16C(\|\boldsymbol{g}_1^i\|_2 + \|\boldsymbol{g}_2^i\|_2)}{\sin^3 \nu} + \frac{12}{\sin^4 \nu} \right) \exp\left( -\frac{1}{2k^2}g_{1i}^2 \cot^2 \nu \right) \right],$$

and the expectation factors over $g_{1i}, \boldsymbol{g}_1^i, \boldsymbol{g}_2^i$, so we can separately compute the $g_{1i}$ integrals first. For the first of the three terms on the RHS of the previous expression, we have

$$\mathbb{E}_{g_{1i}}\left[ \frac{|g_{1i}|^5}{\sin^6 \nu} \exp\left( -\frac{1}{2k^2}g_{1i}^2 \cot^2 \nu \right) \right] = \frac{1}{\sqrt{2\pi k^2}} \int_{\mathbb{R}} \frac{|g|^5}{\sin^6 \nu} \exp\left( -\frac{1}{2k^2}g^2/\sin^2 \nu \right) \mathrm{d}g$$
$$= \frac{1}{\sqrt{2\pi k^2}} \int_{\mathbb{R}} |g|^5 \exp\left( -\frac{1}{2k^2}g^2 \right) \mathrm{d}g,$$
(E.63)

after the change of variables $g \mapsto g \sin \nu$ in the integral, which is valid whenever $0 < \nu < \pi$. To take care of the case where $\nu = 0$ or $\nu = \pi$, we can use the estimate (E.61), valid for $\nu$ sufficiently close to 0 or $\pi$, and the assumption $g_{1i} \neq 0$ for all $i$ to conclude that $\lim_{\nu \searrow 0} q'(\nu, \boldsymbol{g}_1, \boldsymbol{g}_2) = 0$ for any such fixed $(\boldsymbol{g}_1, \boldsymbol{g}_2)$, and by symmetry the analogous result $\lim_{\nu \nearrow \pi} q'(\nu, \boldsymbol{g}_1, \boldsymbol{g}_2) = 0$; and whenever for some $i$ we have $g_{1i} = 0$, we use (E.60) to see that the term in the sum involving $g_{1i}$ poses no problems as $\nu \searrow 0$ or $\nu \nearrow \pi$ because it is identically 0. Returning to the integral (E.63), we have after a change of variables

$$\frac{1}{\sqrt{2\pi k^2}} \int_{\mathbb{R}} |g|^5 \exp\left( -\frac{1}{2k^2}g^2 \right) \mathrm{d}g = \frac{k^5}{\sqrt{2\pi}} \int_{\mathbb{R}} |g|^5 \exp\left( -\frac{1}{2}g^2 \right) \mathrm{d}g = Ck^5,$$

where $C > 0$ is an absolute constant, and where we use Lemma G.11 for the last equality. The remaining two terms can be treated using the same argument: we get

$$\mathbb{E}_{g_{1i}}\left[\frac{|g_{1i}|^3}{\sin^3 \nu} \exp\left(-\frac{1}{2k^2} g_{1i}^2 \cot^2 \nu\right)\right] = C'k^3$$

(after using $|\sin \nu| \leq 1$) and

$$\mathbb{E}_{g_{1i}}\left[\frac{|g_{1i}|^3}{\sin^4 \nu} \exp\left(-\frac{1}{2k^2} g_{1i}^2 \cot^2 \nu\right)\right] = C''k^3$$

for absolute constants $C', C'' > 0$. Combining these estimates gives

$$\mathbb{E}[|q'(\nu, \boldsymbol{g}_1, \boldsymbol{g}_2)|] \leq \frac{C}{n} \sum_{i=1}^n \mathbb{E}_{\boldsymbol{g}_1^i, \boldsymbol{g}_2^i}\left[\|\boldsymbol{g}_1^i\|_2 + \|\boldsymbol{g}_2^i\|_2\right],$$

and using Lemma G.9 (or equivalently Jensen's inequality) gives finally

$$\mathbb{E}[|q'(\nu, \boldsymbol{g}_1, \boldsymbol{g}_2)|] \leq C\sqrt{\frac{n-1}{n}} \leq C.$$

To conclude, we need to show that $\mathbb{E}_{\boldsymbol{g}_2}[q'(\nu, \boldsymbol{g}_1, \boldsymbol{g}_2)]$ is uniformly bounded by a polynomial in $n$ with high probability. For this we start from the estimate (E.60) and apply the argument following that, but with more care in tracking the constants: if $\nu$ is within $\pi/4$ of either $0$ or $\pi$, we can assert

$$|q'(\nu, \boldsymbol{g}_1, \boldsymbol{g}_2)| \leq \frac{C}{k} \sum_{i=1}^n \frac{C_1 k^4}{|g_{1i}|} + C_2 k^3(\|\boldsymbol{g}_1^i\|_2 + \|\boldsymbol{g}_2^i\|_2) + \frac{C_3 k^4}{|g_{1i}|}$$

whenever $g_{1i} \neq 0$ for every $i$ (a set of full measure); and when $\nu$ is within $\pi/4$ of $\pi/2$, we can assert

$$|q'(\nu, \boldsymbol{g}_1, \boldsymbol{g}_2)| \leq \frac{C}{k} \sum_{i=1}^n \frac{C_1' |g_{1i}|^5}{k^2} + C_2' |g_{1i}|^3(\|\boldsymbol{g}_1^i\|_2 + \|\boldsymbol{g}_2^i\|_2) + C_3' |g_{1i}|^3,$$

where $C_i, C_i' > 0$ are absolute constants. By the triangle inequality, independence, and Lemma G.9, when we consider $|\mathbb{E}_{\boldsymbol{g}_2}[q'(\nu, \boldsymbol{g}_1, \boldsymbol{g}_2)]|$, the term $\mathbb{E}[\|\boldsymbol{g}_2^i\|_2]$ is bounded by an absolute constant. Additionally, by Gauss-Lipschitz concentration and Lemma G.9, we have that simultaneously for all $i$ $\|\boldsymbol{g}_1^i\|_2 \leq \|\boldsymbol{g}_1\|_2 \leq 2$ with probability at least $1 - 2e^{-cn}$. Moreover, since $\|\boldsymbol{g}_1\|_\infty \leq \|\boldsymbol{g}_1\|_2$ we also have control of the magnitude of each $|g_{1i}|$ on this event, so with probability at least $1 - 2e^{-cn}$ we have for every $\nu$

$$\left|\mathbb{E}_{\boldsymbol{g}_2}[q'(\nu, \boldsymbol{g}_1, \boldsymbol{g}_2)]\right| \leq \frac{C}{k} \sum_{i=1}^n \frac{C_1 k^4}{|g_{1i}|} + C_2 k^3 + \frac{C_3}{k^2} + C_4$$

for absolute constants $C, C_i > 0$. If $X \sim \mathcal{N}(0,1)$, we have for any $t \geq 0$ that $\mathbb{P}[|X| \geq t] \geq 1 - Ct$, where $C > 0$ is an absolute constant; so if $X_i \sim_{\text{i.i.d.}} \mathcal{N}(0,1)$, we have by independence and if $t$ is less than an absolute constant $\mathbb{P}[\forall i, |X_i| \geq t] \geq (1 - Ct)^n \geq 1 - C'nt$, where the last inequality uses the numerical inequality $e^{-2t} \leq 1 - t \leq e^{-t}$, valid for $0 \leq t \leq \frac{1}{2}$. From this expression, we conclude that when $0 \leq t \leq cn^{-1/2}$ for an absolute constant $c > 0$, we have

$$\mathbb{P}[\forall i \in [n], |g_{1i}| \geq t] \geq 1 - Cn^{3/2}t,$$

so choosing in particular $t = cn^{-(\delta + \frac{3}{2})}$ for any $\delta > 0$, we conclude that $\mathbb{P}[\forall i \in [n], |g_{1i}| \geq cn^{-3/2-\delta}] \geq 1 - C'n^{-\delta}$. Consequently, for any $\delta > 0$ we have with probability at least $1 - C'n^{-\delta} - 2e^{-cn}$

$$\left|\mathbb{E}_{\boldsymbol{g}_2}[q'(\nu, \boldsymbol{g}_1, \boldsymbol{g}_2)]\right| \leq \frac{C}{k} \sum_{i=1}^n C_1 k^4 n^{3/2+\delta} + C_2 k^3 + \frac{C_3}{k^2} + C_4,$$

and since $k = \sqrt{2/n}$, this yields $|\mathbb{E}_{\boldsymbol{g}_2}[q'(\nu, \boldsymbol{g}_1, \boldsymbol{g}_2)]| \leq C_1 n^{1+\delta} + C_2 + C_3 n^{5/2} + C_4 n^{3/2}$ with the same probability. Consequently we can conclude that for any $\delta \geq 3/2$, we have

$$\mathbb{P}\left[\left|\mathbb{E}_{\boldsymbol{g}_2}[\Xi_1(\nu, \boldsymbol{g}_1, \boldsymbol{g}_2)] - \mathbb{E}_{\boldsymbol{g}_1, \boldsymbol{g}_2}[\Xi_1(\nu, \boldsymbol{g}_1, \boldsymbol{g}_2)]\right| \text{ is } C + C'n^{1+\delta}\text{-Lipschitz}\right] \geq 1 - 2e^{-cn} - C''n^{-\delta}.$$

□

**Lemma E.41.** *In the notation of Lemma E.13, if $d \geq 1$, there are absolute constants $c, c'C, C' > 0$ and an absolute constant $K > 0$ such that if $n \geq K$, there is an event with probability at least $1 - Ce^{-cn}$ on which*

$$\forall \nu \in [0, \pi], \; |\Xi_2(\nu, \boldsymbol{g}_1, \boldsymbol{g}_2) - \mathbb{E}[\Xi_2(\nu, \boldsymbol{g}_1, \boldsymbol{g}_2)]| \leq C'e^{-c'n}.$$

*Proof.* Let $\mathcal{E}$ denote the event $\mathcal{E}_{0.5,0}$ in Lemma E.16; then by that lemma, $\mathcal{E}$ has probability at least $1 - Ce^{-cn}$ as long as $n \geq C'$, where $c, C, C' > 0$ are absolute constants, and for $(\boldsymbol{g}_1, \boldsymbol{g}_2) \in \mathcal{E}$, one has for all $\nu \in [0, \pi]$

$$\Xi_2(\nu, \boldsymbol{g}_1, \boldsymbol{g}_2) = 0.$$

This allows us to calculate, for each $\nu$,

$$\mathbb{E}[\Xi_2(\nu, \boldsymbol{g}_1, \boldsymbol{g}_2)] = \mathbb{E}[\mathbb{1}_{\mathcal{E}^c}\Xi_2(\nu, \boldsymbol{g}_1, \boldsymbol{g}_2)] \leq \mathbb{E}[\mathbb{1}_{\mathcal{E}^c}]^{1/2}\|\Xi_2(\nu, \cdot)\|_{L^2} \leq C'e^{-c'n},$$

after applying Lemma E.37 and Lyapunov's inequality and worst-casing constants. We conclude that with probability at least $1 - Ce^{-cn}$

$$\forall \nu \in [0, \pi], \; |\Xi_2(\nu, \boldsymbol{g}_1, \boldsymbol{g}_2) - \mathbb{E}[\Xi_2(\nu, \boldsymbol{g}_1, \boldsymbol{g}_2)]| \leq C'e^{-c'n}.$$

$\square$

**Lemma E.42.** *In the notation of Lemma E.13, if $d \geq 1$, there are absolute constants $c, c'C, C' > 0$ and an absolute constant $K > 0$ such that if $n \geq K$, there is an event with probability at least $1 - Ce^{-cn}$ on which*

$$\forall \nu \in [0, \pi], \; |\Xi_3(\nu, \boldsymbol{g}_1, \boldsymbol{g}_2) - \mathbb{E}[\Xi_3(\nu, \boldsymbol{g}_1, \boldsymbol{g}_2)]| \leq C'e^{-c'n}.$$

*Proof.* The argument is identical to Lemma E.41. Let $\mathcal{E}$ denote the event $\mathcal{E}_{0.5,0}$ in Lemma E.16; then by that lemma, $\mathcal{E}$ has probability at least $1 - Ce^{-cn}$ as long as $n \geq C'$, where $c, C, C' > 0$ are absolute constants, and for $(\boldsymbol{g}_1, \boldsymbol{g}_2) \in \mathcal{E}$, one has for all $\nu \in [0, \pi]$

$$\Xi_3(\nu, \boldsymbol{g}_1, \boldsymbol{g}_2) = 0.$$

This allows us to calculate, for each $\nu$,

$$\mathbb{E}[\Xi_3(\nu, \boldsymbol{g}_1, \boldsymbol{g}_2)] = \mathbb{E}[\mathbb{1}_{\mathcal{E}^c}\Xi_3(\nu, \boldsymbol{g}_1, \boldsymbol{g}_2)] \leq \mathbb{E}[\mathbb{1}_{\mathcal{E}^c}]^{1/2}\|\Xi_3(\nu, \cdot)\|_{L^2} \leq C'e^{-c'n},$$

after applying Lemma E.37 and Lyapunov's inequality and worst-casing constants. We conclude that with probability at least $1 - Ce^{-cn}$

$$\forall \nu \in [0, \pi], \; |\Xi_3(\nu, \boldsymbol{g}_1, \boldsymbol{g}_2) - \mathbb{E}[\Xi_3(\nu, \boldsymbol{g}_1, \boldsymbol{g}_2)]| \leq C'e^{-c'n}.$$

$\square$

**Lemma E.43.** *In the notation of Lemma E.13, if $d \geq 1$, there are absolute constants $c, C, C', C'' > 0$ and absolute constants $K, K' > 0$ such that if $n \geq Kd\log n$ and $d \geq K'$, there is an event with probability at least $1 - Ce^{-cn} - C'n^{-d}$ on which one has*

$$\forall \nu \in [0, \pi], \; |\Xi_4(\nu, \boldsymbol{g}_1, \boldsymbol{g}_2) - \mathbb{E}[\Xi_4(\nu, \boldsymbol{g}_1, \boldsymbol{g}_2)]| \leq C''\sqrt{\frac{d\log n}{n}}.$$

*Proof.* We are going to control the expectation first, showing that it is small; then prove that $|\Xi_4|$ is small uniformly in $\nu$. Let $\mathcal{E}$ denote the event $\mathcal{E}_{0.5,0}$ in Lemma E.16; then by that lemma, $\mathcal{E}$ has probability at least $1 - Ce^{-cn}$ as long as $n \geq C'$, where $c, C, C' > 0$ are absolute constants, and for $(\boldsymbol{g}_1, \boldsymbol{g}_2) \in \mathcal{E}$, one has for all $\nu \in [0, \pi]$

$$\Xi_4(\nu, \boldsymbol{g}_1, \boldsymbol{g}_2) = -2\frac{\langle \boldsymbol{v}_0, \dot{\boldsymbol{v}}_\nu\rangle\langle \boldsymbol{v}_\nu, \dot{\boldsymbol{v}}_\nu\rangle}{\|\boldsymbol{v}_0\|_2\|\boldsymbol{v}_\nu\|_2^3}.$$

Thus, if we write

$$\widetilde{\Xi}_4(\nu, \boldsymbol{g}_1, \boldsymbol{g}_2) = -2\mathbb{1}_{\mathcal{E}}(\boldsymbol{g}_1, \boldsymbol{g}_2)\frac{\langle \boldsymbol{v}_0, \dot{\boldsymbol{v}}_\nu\rangle\langle \boldsymbol{v}_\nu, \dot{\boldsymbol{v}}_\nu\rangle}{\|\boldsymbol{v}_0\|_2\|\boldsymbol{v}_\nu\|_2^3},$$

we have $\widetilde{\Xi}_4 = \Xi_4$ for all $\nu$ whenever $(\boldsymbol{g}_1, \boldsymbol{g}_2) \in \mathcal{E}$, so that for any $\nu$

$$|\mathbb{E}[\Xi_4(\nu, \boldsymbol{g}_1, \boldsymbol{g}_2)]| = \left|\mathbb{E}[\widetilde{\Xi}_4(\nu, \boldsymbol{g}_1, \boldsymbol{g}_2)] + \mathbb{E}[\mathbb{1}_{\mathcal{E}^c}\Xi_4(\nu, \boldsymbol{g}_1, \boldsymbol{g}_2)]\right|$$
$$\leq |\mathbb{E}[\widetilde{\Xi}_4(\nu, \boldsymbol{g}_1, \boldsymbol{g}_2)]| + Ce^{-cn},$$

where the second line uses the triangle inequality and the Schwarz inequality and Lemma E.37 together with the Lyapunov inequality. We proceed with analyzing the expectation of $\widetilde{\Xi}_4$. Using the Schwarz inequality gives

$$\left|\mathbb{E}\left[\widetilde{\Xi}_4(\nu, \boldsymbol{g}_1, \boldsymbol{g}_2)\right]\right| \leq 2\mathbb{E}\left[\langle \boldsymbol{v}_0, \dot{\boldsymbol{v}}_\nu\rangle^2 \langle \boldsymbol{v}_\nu, \dot{\boldsymbol{v}}_\nu\rangle^2\right]^{1/2}\mathbb{E}\left[\frac{\mathbb{1}_{\mathcal{E}}}{\|\boldsymbol{v}_0\|_2^2\|\boldsymbol{v}_\nu\|_2^6}\right]^{1/2}$$
$$\leq 32\mathbb{E}\left[\langle \boldsymbol{v}_0, \dot{\boldsymbol{v}}_\nu\rangle^2\langle \boldsymbol{v}_\nu, \dot{\boldsymbol{v}}_\nu\rangle^2\right]^{1/2},$$

and the checks at and around (E.34) and (E.35) in the proof of Lemma E.15 show that we can apply Lemma E.30 to obtain

$$\left|\begin{array}{c}\mathbb{E}\left[\langle \boldsymbol{v}_0, \dot{\boldsymbol{v}}_\nu\rangle^2\langle \boldsymbol{v}_\nu, \dot{\boldsymbol{v}}_\nu\rangle^2\right]\\ -n^4\left(\begin{array}{c}\mathbb{E}[\sigma(g_{11})\dot{\sigma}(g_{11}\cos\nu + g_{21}\sin\nu)(g_{21}\cos\nu - g_{11}\sin\nu)]^2\\ *\mathbb{E}[\sigma(g_{11}\cos\nu + g_{21}\sin\nu)(g_{21}\cos\nu - g_{11}\sin\nu)]^2\end{array}\right)\end{array}\right| \leq C/n.$$

But we have using rotational invariance that $\mathbb{E}[\sigma(g_{11}\cos\nu + g_{21}\sin\nu)(g_{21}\cos\nu - g_{11}\sin\nu)] = 0$, which implies

$$\left|\mathbb{E}\left[\langle \boldsymbol{v}_0, \dot{\boldsymbol{v}}_\nu\rangle^2\langle \boldsymbol{v}_\nu, \dot{\boldsymbol{v}}_\nu\rangle^2\right]\right| \leq C/n,$$

from which we conclude

$$|\mathbb{E}[\Xi_4(\nu, \boldsymbol{g}_1, \boldsymbol{g}_2)]| \leq C/\sqrt{n}.$$

Next, we control the deviations of $\Xi_4$ with high probability. By Lemma E.17, there is an event $\mathcal{E}_a$ with probability at least $1 - e^{-cn}$ on which $\|\dot{\boldsymbol{v}}_\nu\|_2 \leq 4$ for every $\nu \in [0, \pi]$. Therefore on the event $\mathcal{E}_b = \mathcal{E} \cap \mathcal{E}_a$, which has probability at least $1 - Ce^{-cn}$ by a union bound, we have using Cauchy-Schwarz that for every $\nu$

$$|\Xi_4(\nu, \boldsymbol{g}_1, \boldsymbol{g}_2)| \leq 256|\langle \boldsymbol{v}_\nu, \dot{\boldsymbol{v}}_\nu\rangle|.$$

The coordinates of the random vector $\boldsymbol{v}_\nu \odot \dot{\boldsymbol{v}}_\nu$ are $\sigma(g_{1i}\cos\nu + g_{2i}\sin\nu)(g_{2i}\cos\nu - g_{1i}\sin\nu)$, and we note

$$\mathbb{E}[\sigma(g_{1i}\cos\nu + g_{2i}\sin\nu)(g_{2i}\cos\nu - g_{1i}\sin\nu)] = -\mathbb{E}[\sigma(g_{1i})g_{2i}] = 0,$$

by rotational invariance. Moreover, the calculation (E.35) together with Lemmas G.11 and E.17 demonstrates subexponential moment growth with rate $C/n$, so Lemma G.2 implies for $t \geq 0$

$$\mathbb{P}[\langle \boldsymbol{v}_\nu, \dot{\boldsymbol{v}}_\nu\rangle \geq t] \leq 2e^{-cnt\min\{c't, 1\}}.$$

For large enough $n$, this gives

$$\mathbb{P}\left[\langle \boldsymbol{v}_\nu, \dot{\boldsymbol{v}}_\nu\rangle \geq C\sqrt{\frac{d\log n}{n}}\right] \leq 2n^{-2d}.$$

We turn to the uniformization of this pointwise bound. The map $\nu \mapsto \sum_i \sigma(g_{1i}\cos\nu + g_{2i}\sin\nu)(g_{2i}\cos\nu - g_{1i}\sin\nu)$ is continuous, and differentiable at all but finitely many points of $[0, \pi]$ (following the zero crossings argument in the proof of Lemma E.22) with derivative

$$\nu \mapsto \sum_i \dot{\sigma}(g_{1i}\cos\nu + g_{2i}\sin\nu)(g_{2i}\cos\nu - g_{1i}\sin\nu)^2 - \sigma(g_{1i}\cos\nu + g_{2i}\sin\nu)^2,$$

which is evidently integrable using the triangle inequality and Lemma G.11. In particular, we can write the derivative as $\|\dot{\boldsymbol{v}}_\nu\|_2^2 - \|\boldsymbol{v}_0\|_2^2$. Thus, by (Cohn, 2013, Theorem 6.3.11), to get a Lipschitz estimate on $\nu \mapsto \langle \boldsymbol{v}_\nu, \dot{\boldsymbol{v}}_\nu\rangle$ it suffices to bound the magnitude of the derivative $\nu \mapsto \|\dot{\boldsymbol{v}}_\nu\|_2^2 - \|\boldsymbol{v}_0\|_2^2$. But this is immediate, since on the event $\mathcal{E}_b$ we have $|\|\dot{\boldsymbol{v}}_\nu\|_2^2 - \|\boldsymbol{v}_0\|_2^2| \leq 20$. It thus follows from Lemma E.48 that with probability at least $1 - Ce^{-cn} - C'n^{-2d+1/2}$ we have

$$\forall \nu \in [0, \pi], \ \langle \boldsymbol{v}_\nu, \dot{\boldsymbol{v}}_\nu\rangle \leq C''\sqrt{\frac{d\log n}{n}}. \tag{E.64}$$

As long as $d \geq \frac{1}{2}$, we have that this probability is at least $1 - Ce^{-cn} - C'n^{-d}$, and so the triangle inequality and a union bound yield finally that with probability at least $1 - Ce^{-cn} - C'n^{-d}$

$$\forall \nu \in [0, \pi], \ |\Xi_4(\nu, \boldsymbol{g}_1, \boldsymbol{g}_2) - \mathbb{E}[\Xi_4(\nu, \boldsymbol{g}_1, \boldsymbol{g}_2)]| \leq C'' \sqrt{\frac{d \log n}{n}}.$$

$\square$

**Lemma E.44.** *In the notation of Lemma E.13, if $d \geq 1$, there are absolute constants $c, c', c'', C, C', C'', C''', C''''  > 0$ and an absolute constant $K > 0$ such that if $n \geq Kd \log n$, there is an event with probability at least $1 - Ce^{-cn} + C'e^{-d}$ on which one has*

$$|\Xi_5(\nu, \boldsymbol{g}_1, \boldsymbol{g}_2) - \mathbb{E}[\Xi_5(\nu, \boldsymbol{g}_1, \boldsymbol{g}_2)]| \leq C'' \sqrt{\frac{d}{n}} + C''' e^{-c'd} + C'''' e^{-c''n},$$

*Proof.* Fix $\nu \in [0, \pi]$. Let $\mathcal{E}$ denote the event $\mathcal{E}_{0.5,0}$ in Lemma E.16; then by that lemma, $\mathcal{E}$ has probability at least $1 - Ce^{-cn}$ as long as $n \geq C'$, where $c, C, C' > 0$ are absolute constants, and for $(\boldsymbol{g}_1, \boldsymbol{g}_2) \in \mathcal{E}$, one has for all $\nu \in [0, \pi]$

$$\Xi_5(\nu, \boldsymbol{g}_1, \boldsymbol{g}_2) = -\frac{\langle \boldsymbol{v}_0, \boldsymbol{v}_\nu \rangle \|\dot{\boldsymbol{v}}_\nu\|_2^2}{\|\boldsymbol{v}_0\|_2 \|\boldsymbol{v}_\nu\|_2^3}.$$

Thus, if we write

$$\widetilde{\Xi}_5(\nu, \boldsymbol{g}_1, \boldsymbol{g}_2) = -\mathbb{1}_{\mathcal{E}}(\boldsymbol{g}_1, \boldsymbol{g}_2) \frac{\langle \boldsymbol{v}_0, \boldsymbol{v}_\nu \rangle \|\dot{\boldsymbol{v}}_\nu\|_2^2}{\|\boldsymbol{v}_0\|_2 \|\boldsymbol{v}_\nu\|_2^3}$$

we have $\widetilde{\Xi}_5 = \Xi_5$ for any $\nu$ whenever $(\boldsymbol{g}_1, \boldsymbol{g}_2) \in \mathcal{E}$, so that by the triangle inequality, for any $\nu$

$$\begin{aligned}
|\Xi_5(\nu, \boldsymbol{g}_1, \boldsymbol{g}_2) - \mathbb{E}[\Xi_5(\nu, \boldsymbol{g}_1, \boldsymbol{g}_2)]| &\leq \left| \widetilde{\Xi}_5(\nu, \boldsymbol{g}_1, \boldsymbol{g}_2) - \mathbb{E}\left[\widetilde{\Xi}_5(\nu, \boldsymbol{g}_1, \boldsymbol{g}_2)\right] \right| \\
&\quad + \left| \mathbb{E}[\Xi_5(\nu, \boldsymbol{g}_1, \boldsymbol{g}_2)] - \mathbb{E}\left[\widetilde{\Xi}_5(\nu, \boldsymbol{g}_1, \boldsymbol{g}_2)\right] \right| \\
&\leq \left| \widetilde{\Xi}_5(\nu, \boldsymbol{g}_1, \boldsymbol{g}_2) - \mathbb{E}\left[\widetilde{\Xi}_5(\nu, \boldsymbol{g}_1, \boldsymbol{g}_2)\right] \right| \\
&\quad + \mathbb{E}\left[\mathbb{1}_{\mathcal{E}^c} \left| \Xi_5(\nu, \boldsymbol{g}_1, \boldsymbol{g}_2) - \widetilde{\Xi}_5(\nu, \boldsymbol{g}_1, \boldsymbol{g}_2) \right| \right] \\
&\leq \left| \widetilde{\Xi}_5(\nu, \boldsymbol{g}_1, \boldsymbol{g}_2) - \mathbb{E}\left[\widetilde{\Xi}_5(\nu, \boldsymbol{g}_1, \boldsymbol{g}_2)\right] \right| + Ce^{-cn},
\end{aligned}$$

where the second line uses the triangle inequality, and the third line uses the Schwarz inequality and Lemma E.37 together with the Lyapunov inequality.

So, we can proceed analyzing $\widetilde{\Xi}_5$. First, we aim to apply Lemma E.33 with the choices

$$X = -\mathbb{1}_{\mathcal{E}} \frac{\langle \boldsymbol{v}_0, \boldsymbol{v}_\nu \rangle}{\|\boldsymbol{v}_0\|_2 \|\boldsymbol{v}_\nu\|_2}; \qquad Y = \mathbb{1}_{\mathcal{E}} \frac{\|\dot{\boldsymbol{v}}_\nu\|_2^2}{\|\boldsymbol{v}_\nu\|_2^2},$$

since $XY = \widetilde{\Xi}_5(\nu, \cdot, \cdot)$; square-integrability of $X$ and $Y$ is evident from the definition of $\mathbb{1}_{\mathcal{E}}$, and we have $|X| \leq 1$ by Cauchy-Schwarz. To control $Y$, we start by noting

$$\|Y - 1\|_{L^2} \leq 1 + \|Y\|_{L^2} \leq 1 + 4\mathbb{E}\left[\|\dot{\boldsymbol{v}}_\nu\|_2^4\right]^{1/2} \leq 1 + 4\sqrt{1 + C},$$

where $C > 0$ is an absolute constant; the first inequality is the Minkowski inequality, the second uses the property of $\mathcal{E}$ and drops the indicator by nonnegativity, and the third applies Lemma E.29, and discards the $n^{-1}$ factor. For deviations, we start by noting that $\mathbb{E}[\|\boldsymbol{v}_\nu\|_2^2] = 1$, and that by Lemmas G.2 and G.11, we have

$$\mathbb{P}\left[\left| \|\boldsymbol{v}_\nu\|_2^2 - 1 \right| \geq t\right] \leq 2e^{-cnt \min\{Ct, 1\}}.$$

It follows that there exists an absolute constant $C' > 0$ such that, putting $t = C'\sqrt{d/n}$ and choosing $n \geq (C'/C)^2 d$, we have

$$\mathbb{P}\left[\left| \|\boldsymbol{v}_\nu\|_2^2 - 1 \right| \geq C'\sqrt{\frac{d}{n}}\right] \leq 2e^{-d}. \tag{E.65}$$

Moreover, by Lemma E.17, we can run a similar argument on $\|\dot{\boldsymbol{v}}_\nu\|_2^2$ to get that if $n$ is larger than a constant multiple of $d$

$$\mathbb{P}\left[\left|\|\dot{\boldsymbol{v}}_\nu\|_2^2 - 1\right| \geq C\sqrt{\frac{d}{n}}\right] \leq 2e^{-d}. \tag{E.66}$$

Next, Taylor expansion with Lagrange remainder of the smooth function $x \mapsto x^{-1}$ on the domain $x > 0$ about the point 1 gives

$$\frac{1}{x} = 1 - (x-1) + \frac{1}{\xi^3}(x-1)^2, \tag{E.67}$$

where $\xi$ lies between 1 and $x$. If $(\boldsymbol{g}_1, \boldsymbol{g}_2) \in \mathcal{E}$, then $\|\boldsymbol{v}_\nu\|_2^6 \geq (1/64)$, and we can therefore assert

$$1 - \left(\|\boldsymbol{v}_\nu\|_2^2 - 1\right) \leq \frac{1}{\|\boldsymbol{v}_\nu\|_2^2} \leq 1 - \left(\|\boldsymbol{v}_\nu\|_2^2 - 1\right) + 64\left(\|\boldsymbol{v}_\nu\|_2^2 - 1\right)^2$$

with probability at least $1 - Ce^{-cn}$. Using a union bound together with (E.65) (and changing the constant to $C$), we have with probability at least $1 - 2e^{-cd} - C'e^{-c'n}$ that

$$-C\sqrt{\frac{d}{n}} - 64C^2\frac{d}{n} \leq 1 - \frac{1}{\|\boldsymbol{v}_\nu\|_2^2} \leq C\sqrt{\frac{d}{n}}.$$

Given that $n \geq d$, it follows that with the same probability we have

$$-C(1 + 64C)\sqrt{\frac{d}{n}} \leq 1 - \frac{1}{\|\boldsymbol{v}_\nu\|_2^2} \leq C\sqrt{\frac{d}{n}},$$

which implies that with probability at least $1 - 2e^{-d} - C'e^{-cn}$, we have

$$\left|1 - \frac{1}{\|\boldsymbol{v}_\nu\|_2^2}\right| \leq C\sqrt{\frac{d}{n}}.$$

Now, the triangle inequality gives

$$\left|\frac{\|\dot{\boldsymbol{v}}_\nu\|_2^2}{\|\boldsymbol{v}_\nu\|_2^2} - 1\right| \leq \left|\frac{\|\dot{\boldsymbol{v}}_\nu\|_2^2}{\|\boldsymbol{v}_\nu\|_2^2} - \frac{1}{\|\boldsymbol{v}_\nu\|_2^2}\right| + \left|\frac{1}{\|\boldsymbol{v}_\nu\|_2^2} - 1\right|$$

$$\leq \frac{1}{\|\boldsymbol{v}_\nu\|_2^2}\left|\|\dot{\boldsymbol{v}}_\nu\|_2^2 - 1\right| + \left|\frac{1}{\|\boldsymbol{v}_\nu\|_2^2} - 1\right|.$$

When $(\boldsymbol{g}_1, \boldsymbol{g}_2) \in \mathcal{E}$, we have $\|\boldsymbol{v}_\nu\|_2^2 \geq \frac{1}{4}$, so, by a union bound, with probability at least $1 - 4e^{-d} - C'e^{-cn}$ we have

$$\left|\frac{\|\dot{\boldsymbol{v}}_\nu\|_2^2}{\|\boldsymbol{v}_\nu\|_2^2} - 1\right| \leq 4C\sqrt{\frac{d}{n}},$$

and since

$$(\boldsymbol{g}_1, \boldsymbol{g}_2) \in \mathcal{E} \implies Y = \frac{\|\dot{\boldsymbol{v}}_\nu\|_2^2}{\|\boldsymbol{v}_\nu\|_2^2},$$

another union bound and the measure bound on $\mathcal{E}$ let us conclude that with probability at least $1 - 4e^{-d} - C'e^{-cn}$, we have

$$|Y - 1| \leq 4C\sqrt{\frac{d}{n}}.$$

If we choose $n \geq (1/c)(d + \log C'/4)$, we have $4e^{-d} + C'e^{-cn} \leq 8e^{-d}$, so the previous bound occurs with probability at least $1 - 8e^{-d}$. We can now apply Lemma E.33 to get with probability at least $1 - 8e^{-d}$

$$\left|\widetilde{\Xi}_5 - \mathbb{E}\left[\widetilde{\Xi}_5\right]\right| \leq \left|\mathbb{1}_{\mathcal{E}}\left\langle\frac{\boldsymbol{v}_0}{\|\boldsymbol{v}_0\|_2}, \frac{\boldsymbol{v}_\nu}{\|\boldsymbol{v}_\nu\|_2}\right\rangle - \mathbb{E}\left[\mathbb{1}_{\mathcal{E}}\left\langle\frac{\boldsymbol{v}_0}{\|\boldsymbol{v}_0\|_2}, \frac{\boldsymbol{v}_\nu}{\|\boldsymbol{v}_\nu\|_2}\right\rangle\right]\right| + C\sqrt{\frac{d}{n}} + C'e^{-d/2}.$$

Next, we attempt to apply Lemma E.33 again, this time to $X = \mathbb{1}_{\mathcal{E}}\langle\boldsymbol{v}_0, \boldsymbol{v}_\nu\rangle$ and $Y = \mathbb{1}_{\mathcal{E}}(\|\boldsymbol{v}_0\|_2\|\boldsymbol{v}_\nu\|_2)^{-1}$. Using the definition of $\mathcal{E}$, we have $|X| \leq 4$ and $\|Y - 1\|_{L^2} \leq \|Y\|_{L^2} + 1 \leq 5$,

where the second bound also leverages the Minkowski inequality; so we need only establish deviations of $Y$. Applying again (E.67), and using $(g_1, g_2) \in \mathcal{E}$ implies $\|v_0\|_2 \|v_\nu\|_2 \geq \frac{1}{4}$, we get

$$(\|v_0\|_2 \|v_\nu\|_2 - 1) - 64 \left(\|v_0\|_2 \|v_\nu\|_2 - 1\right)^2 \leq 1 - \frac{1}{\|v_0\|_2 \|v_\nu\|_2} \leq (\|v_0\|_2 \|v_\nu\|_2 - 1) \quad \text{(E.68)}$$

with probability at least $1 - Ce^{-cn}$. Using Lemma G.11 and (Vershynin, 2018, Theorem 3.1.1), we can assert for any $\nu \in [0, \pi]$ and any $t \geq 0$

$$\mathbb{P}[|\|v_\nu\|_2 - 1| \geq t] \leq 2e^{-cnt^2},$$

which implies that there exists an absolute constant $C > 0$ such that for any $d > 0$

$$\mathbb{P}\left[ |\|v_\nu\|_2 - 1| \geq C\sqrt{\frac{d}{n}} \right] \leq 2e^{-d}.$$

In particular, when $n \geq d$, we can assert that $\|v_\nu\|_2 \leq 1 + C$ with probability at least $1 - 2e^{-d}$. By the triangle inequality and a union bound, it follows

$$|\|v_0\|_2 \|v_\nu\|_2 - 1| \leq \|v_0\|_2 |\|v_\nu\|_2 - 1| + |\|v_0\|_2 - 1|$$

$$\leq C\sqrt{\frac{d}{n}}$$

with probability at least $1 - 6e^{-d}$. Then a union bound gives that with probability at least $1 - 6e^{-d} - C'e^{-cn}$, (E.68) leads to

$$-C\sqrt{\frac{d}{n}} \left( 1 + 64C\sqrt{\frac{d}{n}} \right) \leq 1 - \frac{1}{\|v_0\|_2 \|v_\nu\|_2} \leq C\sqrt{\frac{d}{n}},$$

and using $n \geq d$ and worst-casing constants implies that with the same probability

$$\left| 1 - \frac{1}{\|v_0\|_2 \|v_\nu\|_2} \right| \leq C\sqrt{\frac{d}{n}}.$$

Then since $(g_1, g_2) \in \mathcal{E} \implies Y = (\|v_0\|_2 \|v_\nu\|_2)^{-1}$, another union bound gives that with probability at least $1 - 6e^{-d} - C'e^{-cn}$ we have $|Y - 1| \leq C\sqrt{d/n}$. As in the previous step of the reduction, we can choose $n \geq (1/c)(d + \log C'/6)$ to get that $6e^{-d} + C'e^{-cn} \leq 12e^{-d}$, so that the previous bound occurs with probability at least $1 - 12e^{-d}$. We can thus apply Lemma E.33, a union bound, and our previous work to get that with probability at least $1 - 20e^{-d}$

$$\left| \widetilde{\Xi}_5 - \mathbb{E}\left[ \widetilde{\Xi}_5 \right] \right| \leq |\mathbb{1}_\mathcal{E} \langle v_0, v_\nu \rangle - \mathbb{E}[\mathbb{1}_\mathcal{E} \langle v_0, v_\nu \rangle]| + C\sqrt{\frac{d}{n}} + C'e^{-d/2}.$$

Whenever $(g_1, g_2) \in \mathcal{E}$, we have by the triangle inequality, the Schwarz inequality, and Lemmas E.16 and E.29 that

$$|\mathbb{1}_\mathcal{E} \langle v_0, v_\nu \rangle - \mathbb{E}[\mathbb{1}_\mathcal{E} \langle v_0, v_\nu \rangle]| \leq |\langle v_0, v_\nu \rangle - \mathbb{E}[\langle v_0, v_\nu \rangle]| + |\mathbb{E}[\langle v_0, v_\nu \rangle] - \mathbb{E}[\mathbb{1}_\mathcal{E} \langle v_0, v_\nu \rangle]|$$

$$\leq |\langle v_0, v_\nu \rangle - \mathbb{E}[\langle v_0, v_\nu \rangle]| + Ce^{-cn},$$

allowing us to drop the indicator. We have $\langle v_0, v_\nu \rangle = \sum_i \sigma(g_{1i}) \sigma(g_{1i} \cos \nu + g_{2i} \sin \nu)$, which is a sum of independent random variables; following the argument at and around (E.36), we conclude moreover that these random variables are subexponential with rate $C/n$, where $C > 0$ is an absolute constant. We therefore obtain from Lemma G.2 the tail bound

$$\mathbb{P}[|\langle v_0, v_\nu \rangle - \mathbb{E}[\langle v_0, v_\nu \rangle]| \geq t] \leq 2e^{-cnt \min\{Ct, 1\}},$$

which, for a suitable choice of absolute constant $C' > 0$ and choosing $n \geq C'd$, yields the deviations bounds

$$\mathbb{P}\left[ |\langle v_0, v_\nu \rangle - \mathbb{E}[\langle v_0, v_\nu \rangle]| \geq C\sqrt{\frac{d}{n}} \right] \leq 2e^{-d}.$$

Taking a final union bound (since we assumed throughout that $(\boldsymbol{g}_1, \boldsymbol{g}_2) \in \mathcal{E}$) gives that with probability at least $1 - Ce^{-cn} + C'e^{-d}$, one has

$$|\Xi_5(\nu, \boldsymbol{g}_1, \boldsymbol{g}_2) - \mathbb{E}[\Xi_5(\nu, \boldsymbol{g}_1, \boldsymbol{g}_2)]| \le C'' \sqrt{\frac{d}{n}} + C''' e^{-c'd} + C'''' e^{-c''n},$$

which is sufficient to conclude pointwise concentration as claimed for sufficiently large $n$ after we put $d = d' \log n$ and include extra $\log n$ factors in any points where we need to choose $n$ larger than $d$. $\qquad\square$

**Lemma E.45.** *In the notation of Lemma E.13, there are absolute constants $c, C, C', C'', C''' > 0$ such that for any $\delta \ge \frac{1}{2}$, one has*

$$\mathbb{P}\left[\left|\mathbb{E}_{\boldsymbol{g}_2}[\Xi_5(\nu, \boldsymbol{g}_1, \boldsymbol{g}_2)] - \mathbb{E}_{\boldsymbol{g}_1, \boldsymbol{g}_2}[\Xi_5(\nu, \boldsymbol{g}_1, \boldsymbol{g}_2)]\right| \text{ is } C + C'n^{1+\delta}\text{-Lipschitz}\right] \ge 1 - C''e^{-cn} - C'''n^{-\delta}$$

*as long as $\delta \ge \frac{1}{2}$.*

*Proof.* We will differentiate with respect to $\nu$ the function

$$f(\nu, \boldsymbol{g}_1) = -\mathbb{E}_{\boldsymbol{g}_2}\left[\frac{\langle \boldsymbol{v}_0, \boldsymbol{v}_\nu \rangle \|\dot{\boldsymbol{v}}_\nu\|_2^2 \psi'(\|\boldsymbol{v}_\nu\|_2)}{\psi(\|\boldsymbol{v}_0\|_2)\psi(\|\boldsymbol{v}_\nu\|_2)^2\|\boldsymbol{v}_\nu\|_2}\right],$$

and construct an event on which $f'$ has size $\mathrm{poly}(n)$. We need to also differentiate the function $\mathbb{E}[f(\cdot, \boldsymbol{g}_1)]$; for this we will additionally show that $f'(\nu, \cdot)$ is absolutely integrable over the product $[0, \pi] \times \mathbb{R}^n \times \mathbb{R}^n$, which allows us to apply Fubini's theorem to move both the $\boldsymbol{g}_1$ and $\boldsymbol{g}_2$ expectations under the $\nu$ integral in the first-order Taylor expansion we obtain. In particular, the derivative of $\mathbb{E}[f(\cdot, \boldsymbol{g}_1)]$ will in this way be shown to be $\mathbb{E}[f'(\cdot, \boldsymbol{g}_1)]$, so that linearity and the triangle inequality imply a $\mathrm{poly}(n)$ magnitude bound for the derivative of $\mathbb{E}_{\boldsymbol{g}_2}[\Xi_5] - \mathbb{E}[\Xi_5]$.

Define

$$q_i(\nu, \boldsymbol{g}_1, \boldsymbol{g}_2) = \frac{\langle \boldsymbol{v}_0, \boldsymbol{v}_\nu \rangle \psi'(\|\boldsymbol{v}_\nu\|_2)(g_{2i}\cos\nu - g_{1i}\sin\nu)^2}{\psi(\|\boldsymbol{v}_0\|_2)\psi(\|\boldsymbol{v}_\nu\|_2)^2\|\boldsymbol{v}_\nu\|_2},$$

so that, for almost all $\boldsymbol{g}_1$,

$$f(\nu, \boldsymbol{g}_1) = -\sum_{i=1}^{n} \mathbb{E}_{\boldsymbol{g}_2}[q_i(\nu, \boldsymbol{g}_1, \boldsymbol{g}_2)\dot{\sigma}(g_{1i}\cos\nu + g_{2i}\sin\nu)].$$

For each fixed $(\boldsymbol{g}_1, \boldsymbol{g}_2)$ and each $i$, the only obstructions to differentiability of $q_i$ in $\nu$ arise from the function $\sigma$ (using smoothness of $\psi$ from Lemma E.31 and the fact that it is constant whenever $\|\boldsymbol{v}_\nu\|$ is small enough that nondifferentiability of $\|\cdot\|_2$ could pose a problem); following the zero-crossings argument of Lemma E.22, $q_i$ fails to be differentiable at no more than $n$ points of $[0, \pi]$, and otherwise has derivative

$$q_i'(\nu, \boldsymbol{g}_1, \boldsymbol{g}_2) =$$
$$\frac{1}{\psi(\|\boldsymbol{v}_0\|_2)}\left(\frac{\langle \boldsymbol{v}_0, \dot{\boldsymbol{v}}_\nu \rangle \psi'(\|\boldsymbol{v}_\nu\|_2)(g_{2i}\cos\nu - g_{1i}\sin\nu)^2}{\psi(\|\boldsymbol{v}_\nu\|_2)^2\|\boldsymbol{v}_\nu\|_2}\right.$$
$$+ \frac{\langle \boldsymbol{v}_0, \boldsymbol{v}_\nu \rangle \langle \boldsymbol{v}_\nu, \dot{\boldsymbol{v}}_\nu \rangle \psi''(\|\boldsymbol{v}_\nu\|_2)(g_{2i}\cos\nu - g_{1i}\sin\nu)^2}{\psi(\|\boldsymbol{v}_\nu\|_2)^2\|\boldsymbol{v}_\nu\|_2^2}$$
$$- 2\frac{\langle \boldsymbol{v}_0, \boldsymbol{v}_\nu \rangle \psi'(\|\boldsymbol{v}_\nu\|_2)(g_{2i}\cos\nu - g_{1i}\sin\nu)(g_{1i}\cos\nu + g_{2i}\sin\nu)}{\psi(\|\boldsymbol{v}_\nu\|_2)^2\|\boldsymbol{v}_\nu\|_2}$$
$$- 2\frac{\psi'(\|\boldsymbol{v}_\nu\|_2)^2\langle \boldsymbol{v}_\nu, \dot{\boldsymbol{v}}_\nu \rangle \langle \boldsymbol{v}_0, \boldsymbol{v}_\nu \rangle(g_{2i}\cos\nu - g_{1i}\sin\nu)^2}{\psi(\|\boldsymbol{v}_\nu\|_2)^3\|\boldsymbol{v}_\nu\|_2^2}$$
$$\left.- \frac{\psi'(\|\boldsymbol{v}_\nu\|_2)\langle \boldsymbol{v}_\nu, \dot{\boldsymbol{v}}_\nu \rangle \langle \boldsymbol{v}_0, \boldsymbol{v}_\nu \rangle(g_{2i}\cos\nu - g_{1i}\sin\nu)^2}{\psi(\|\boldsymbol{v}_\nu\|_2)^2\|\boldsymbol{v}_\nu\|_2^3}\right). \tag{E.69}$$

by the chain rule and the product rule. To conclude absolute continuity of $q_i(\cdot, \boldsymbol{g}_1, \boldsymbol{g}_2)$, we need to show that $q_i'$ is integrable; this follows from Cauchy-Schwarz, the integrability of $\|\boldsymbol{v}_0\|_2$, $\|\boldsymbol{v}_\nu\|_2$,

$\|\dot{\boldsymbol{v}}_\nu\|_2$ (Lemma E.17), the triangle inequality, and the Lemma E.31 estimates $\psi \geq \frac{1}{4}$, $|\psi'| \leq C$, $|\psi''| \leq C'$, and $|\psi'(x)/x| \leq C''$ for any $x \in \mathbb{R}$ (to see this last estimate, note that $|\psi'|$ is bounded on $\mathbb{R}$, and use that $\psi$ is constant whenever $x \leq \frac{1}{4}$). Then (Cohn, 2013, Theorem 6.3.11) implies that $q_i(\,\cdot\,, \boldsymbol{g}_1, \boldsymbol{g}_2)$ is absolutely continuous with a.e. derivative $q_i'$. Next, we can write

$$f(\nu, \boldsymbol{g}_1) = -\sum_{i=1}^n \mathop{\mathbb{E}}_{g_{2j}:j\neq i}\left[\mathop{\mathbb{E}}_{g_{2i}}[q_i(\nu, \boldsymbol{g}_1, \boldsymbol{g}_2)\dot{\sigma}(g_{1i}\cos\nu + g_{2i}\sin\nu)]\right],$$

using Lemma E.37 to see that Fubini's theorem can be applied. Our aim is now to apply Lemma E.27, so we need to check its remaining hypotheses. First, continuity of $q_i(\nu, \,\cdot\,)$ follows from continuity of $\sigma$, smoothness of $\psi$, and the fact that the denominator never vanishes. Joint absolute integrability of $q_i$ and $q_i'$ follows from our verification of absolute integrability of $q_i'$ above, which produces a final upper bound that does not depend on $\nu$ (which is therefore integrable over $[0, \pi]$ as well); the corresponding result for $q_i$ follows from Lemma E.37. Last, we need the growth estimate. We have from Lemma E.31

$$|q_i(\nu, \boldsymbol{g}_1, \boldsymbol{g}_2)| \leq 32C(g_{2i}\cos\nu - g_{1i}\sin\nu)^2 \leq 32C(|g_{2i}| + |g_{1i}|)^2 \leq 32C|g_{1i}|(1 + |g_{2i}|)^2,$$

which is evidently quadratic in $|g_{2i}|$ once $|g_{2i}| \geq 1$. Consequently we can apply Lemma E.27 to differentiate $f(\,\cdot\,, \boldsymbol{g}_1)$; we get at almost all $\boldsymbol{g}_1$

$$f(\nu, \boldsymbol{g}_1) = -\sum_{i=1}^n \left( \mathop{\mathbb{E}}_{g_{2j}:j\neq i}\left[\mathop{\mathbb{E}}_{g_{2i}}[q_i(0, \boldsymbol{g}_1, \boldsymbol{g}_2)\dot{\sigma}(g_{1i})]\right] \right.$$
$$\left. + \mathop{\mathbb{E}}_{g_{2j}:j\neq i}\left[\int_0^\nu \mathrm{d}t \left( \begin{array}{c} \mathbb{E}_{g_{2i}}[q_i'(t, \boldsymbol{g}_1, \boldsymbol{g}_2)\dot{\sigma}(g_{1i}\cos t + g_{2i}\sin t)] \\ -g_{1i}q_i(t, \boldsymbol{g}_1, \widetilde{\boldsymbol{g}}_2^i)\rho(-g_{1i}\cot t)\sin^{-2} t \end{array} \right)\right] \right),$$

where $\widetilde{\boldsymbol{g}}_2^i$ is the vector $\boldsymbol{g}_2$ but with its $i$-th coordinate replaced by $-g_{1i}\cot t$, and where $\rho$ is the pdf of a $\mathcal{N}(0, 2/n)$ random variable. The changes in $\widetilde{\boldsymbol{g}}_2^i$ drive updates to the terms in $q_i$ as follows: we have $\sigma(g_{1i}\cos\nu + g_{2i}\sin\nu)$ becoming 0, and $g_{2i}\cos\nu - g_{1i}\sin\nu$ becoming $-g_{1i}/\sin\nu$. Thus, we have

$$-g_{1i}q_i(t, \boldsymbol{g}_1, \widetilde{\boldsymbol{g}}_2^i)\rho(-g_{1i}\cot t)\sin^{-2} t = -\frac{g_{1i}^3\langle \boldsymbol{v}_0^i, \boldsymbol{v}_t^i\rangle\psi'(\|\boldsymbol{v}_t^i\|_2)\rho(-g_{1i}\cot t)}{\psi(\|\boldsymbol{v}_0\|_2)\psi(\|\boldsymbol{v}_t^i\|_2)^2\|\boldsymbol{v}_t^i\|_2\sin^4 t},$$

where the notation $\boldsymbol{v}_t^i$ is in use in the $\Xi_1$ control section and is defined in Lemma E.26, and $\boldsymbol{v}_0^i$ is defined here similarly (the $\mathbb{R}^{n-1}$ vector which is the projection of $\boldsymbol{v}_0$ onto all but the $i$-th coordinates). Using Lemma E.31, we can further assert

$$\left|g_{1i}q_i(t, \boldsymbol{g}_1, \widetilde{\boldsymbol{g}}_2^i)\rho(-g_{1i}\cot t)\sin^{-2} t\right| \leq 16C\frac{|g_{1i}|^3\rho(-g_{1i}\cot t)}{\sin^4 t} \qquad (E.70)$$

where we use that $\|\boldsymbol{v}_0^i\|_2 \leq \|\boldsymbol{v}_0\|_2$. For each fixed $\boldsymbol{g}_1$ having no coordinates equal to zero, we write $K_i = |g_{1i}| > 0$; if $0 \leq t \leq \pi/4$ or $3\pi/4 \leq t \leq \pi$, we have $\cos^2 t \geq \frac{1}{2}$, and so

$$\frac{\rho(-g_{1i}\cot t)}{\sin^4 t} \leq \sqrt{\frac{n}{4\pi}}\sin^{-4} t\exp\left(\frac{K_i^2 n}{8}\frac{1}{\sin^2 t}\right).$$

Using Lemma E.36, we have

$$\frac{\rho(-g_{1i}\cot t)}{\sin^4 t} \leq \sqrt{\frac{n}{4\pi}}\left(\frac{16}{K_i^2 n}\right)^2.$$

On the other hand, when $\pi/4 \leq t \leq 3\pi/4$, then $\sin t \geq 2^{-1/2}$, and we can assert

$$\frac{\rho(-g_{1i}\cot t)}{\sin^4 t} \leq 8\sqrt{n/\pi}.$$

We conclude for any $t$

$$\left|g_{1i}q_i(t, \boldsymbol{g}_1, \widetilde{\boldsymbol{g}}_2^i)\rho(-g_{1i}\cot t)\sin^{-2} t\right| \leq C/(K_i n^{3/2}) + C'\sqrt{n}K_i^3 \qquad (E.71)$$

for absolute constants $C, C' > 0$, and this upper bound is integrable jointly over $t$ and $\boldsymbol{g}_2$. We have checked previously the joint integrability of the $q'_i$ terms when applying Lemma E.27, so we can therefore apply Fubini's theorem to get $\boldsymbol{g}_1$-a.s.

$$f(\nu, \boldsymbol{g}_1) = -\underset{\boldsymbol{g}_2}{\mathbb{E}}\left[\sum_{i=1}^n q_i(0, \boldsymbol{g}_1, \boldsymbol{g}_2)\dot{\sigma}(g_{1i})\right]$$
$$-\int_0^\nu \sum_{i=1}^n \underset{\boldsymbol{g}_2}{\mathbb{E}}\left[\begin{array}{c} q'_i(t, \boldsymbol{g}_1, \boldsymbol{g}_2)\dot{\sigma}(g_{1i}\cos t + g_{2i}\sin t) \\ -g_{1i}q_i(t, \boldsymbol{g}_1, \widetilde{\boldsymbol{g}}_2^i)\rho(-g_{1i}\cot t)\sin^{-2} t \end{array}\right] \mathrm{d}t.$$

Consequently, to conclude a Lipschitz estimate for $f(\,\cdot\,, \boldsymbol{g}_1)$ it suffices to control the quantity under the $t$ integral in the previous expression. We will start by controlling the second term using Markov's inequality. Following (E.70), we calculate

$$\underset{\boldsymbol{g}_1, \boldsymbol{g}_2}{\mathbb{E}}\left[\left|g_{1i}q_i(t, \boldsymbol{g}_1, \widetilde{\boldsymbol{g}}_2^i)\rho(-g_{1i}\cot t)\sin^{-2} t\right|\right] \leq 8C\sqrt{\frac{n}{\pi}}\underset{\boldsymbol{g}_1}{\mathbb{E}}\left[\frac{|g_{1i}|^3 \exp\left(-\frac{n}{4}\frac{g_{1i}^2 \cos^2 t}{\sin^2 t}\right)}{\sin^4 t}\right]$$
$$= \frac{4Cn}{\pi}\int_{\mathbb{R}} \frac{|g|^3}{\sin^4 t}\exp\left(-\frac{n}{4}\frac{g^2}{\sin^2 t}\right)\mathrm{d}g$$
$$= \frac{4Cn}{\pi}\int_{\mathbb{R}}|g|^3 \exp\left(-\frac{n}{4}g^2\right)\mathrm{d}g,$$

where the last line follows from the change of variables $g \mapsto g\sin t$ in the integral. We can evaluate this integral with Lemma G.11, which gives a bound

$$\underset{\boldsymbol{g}_1, \boldsymbol{g}_2}{\mathbb{E}}\left[\left|g_{1i}q_i(t, \boldsymbol{g}_1, \widetilde{\boldsymbol{g}}_2^i)\rho(-g_{1i}\cot t)\sin^{-2} t\right|\right] \leq \frac{128C}{\pi n},$$

and therefore a bound of $C' > 0$ an absolute constant on the sum over $i$. As a byproduct of this estimate, we can assert that the second term is jointly integrable over $[0, \pi] \times \mathbb{R}^n \times \mathbb{R}^n$, which allows us to apply Fubini's theorem and obtain the same differentiation result for $\mathbb{E}[f(\,\cdot\,, \boldsymbol{g}_1)]$. Meanwhile, beginning from (E.71), we can write using the triangle inequality

$$\left|\underset{\boldsymbol{g}_2}{\mathbb{E}}\left[\sum_{i=1}^n g_{1i}q_i(t, \boldsymbol{g}_1, \widetilde{\boldsymbol{g}}_2^i)\rho(-g_{1i}\cot t)\sin^{-2} t\right]\right| \leq \sum_{i=1}^n \frac{C}{|g_{1i}|n^{3/2}} + C'\sqrt{n}|g_{1i}|^3.$$

By Gauss-Lipschitz concentration and Lemma G.9, we have that $\|\boldsymbol{g}_1\|_2 \leq \|\boldsymbol{g}_1\|_2 \leq 2$ with probability at least $1 - 2e^{-cn}$, and since $\|\boldsymbol{g}_1\|_\infty \leq \|\boldsymbol{g}_1\|_2$, we conclude with the same probability that $|g_{1i}| \leq 2$ simultaneously for all $i$. Meanwhile, if $X \sim \mathcal{N}(0, 1)$, we have for any $t \geq 0$ that $\mathbb{P}[|X| \geq t] \geq 1 - Ct$, where $C > 0$ is an absolute constant; so if $X_i \sim_{\text{i.i.d.}} \mathcal{N}(0, 1)$, we have by independence and if $t$ is less than an absolute constant $\mathbb{P}[\forall i, |X_i| \geq t] \geq (1 - Ct)^n \geq 1 - C'nt$, where the last inequality uses the numerical inequality $e^{-2t} \leq 1 - t \leq e^{-t}$, valid for $0 \leq t \leq \frac{1}{2}$. From this expression, we conclude that when $0 \leq t \leq cn^{-1/2}$ for an absolute constant $c > 0$, we have

$$\mathbb{P}[\forall i \in [n], |g_{1i}| \geq t] \geq 1 - Cn^{3/2}t,$$

so choosing in particular $t = cn^{-(\delta+\frac{3}{2})}$ for any $\delta > 0$, we conclude that $\mathbb{P}[\forall i \in [n], |g_{1i}| \geq cn^{-3/2-\delta}] \geq 1 - C'n^{-\delta}$. Then with probability at least $1 - C'n^{-\delta} - 2e^{-cn}$, we have

$$\left|\underset{\boldsymbol{g}_2}{\mathbb{E}}\left[\sum_{i=1}^n g_{1i}q_i(t, \boldsymbol{g}_1, \widetilde{\boldsymbol{g}}_2^i)\rho(-g_{1i}\cot t)\sin^{-2} t\right]\right| \leq Cn^{1+\delta} + C'n^{3/2},$$

so as long as $\delta \geq \frac{1}{2}$, we have

$$\mathbb{P}\left[\left|\underset{\boldsymbol{g}_2}{\mathbb{E}}\left[\sum_{i=1}^n g_{1i}q_i(t, \boldsymbol{g}_1, \widetilde{\boldsymbol{g}}_2^i)\rho(-g_{1i}\cot t)\sin^{-2} t\right]\right| \geq Cn^{1+\delta}\right] \leq C'n^{-\delta} + 2e^{-cn}.$$

Proceeding now to the $q_i'$ term, from the expression (E.69) we get

$$\sum_{i=1}^{n} q_i'(\nu, \boldsymbol{g}_1, \boldsymbol{g}_2) \dot{\sigma}(g_{1i} \cos \nu + g_{2i} \sin \nu)$$

$$= \frac{1}{\psi(\|\boldsymbol{v}_0\|_2)} \left( \frac{\langle \boldsymbol{v}_0, \dot{\boldsymbol{v}}_\nu \rangle \psi'(\|\boldsymbol{v}_\nu\|_2) \|\dot{\boldsymbol{v}}_\nu\|_2^2}{\psi(\|\boldsymbol{v}_\nu\|_2)^2 \|\boldsymbol{v}_\nu\|_2} + \frac{\langle \boldsymbol{v}_0, \boldsymbol{v}_\nu \rangle \langle \boldsymbol{v}_\nu, \dot{\boldsymbol{v}}_\nu \rangle \psi''(\|\boldsymbol{v}_\nu\|_2) \|\dot{\boldsymbol{v}}_\nu\|_2^2}{\psi(\|\boldsymbol{v}_\nu\|_2)^2 \|\boldsymbol{v}_\nu\|_2^2} \right.$$

$$-2 \frac{\langle \boldsymbol{v}_0, \boldsymbol{v}_\nu \rangle \psi'(\|\boldsymbol{v}_\nu\|_2) \langle \dot{\boldsymbol{v}}_\nu, \boldsymbol{v}_\nu \rangle}{\psi(\|\boldsymbol{v}_\nu\|_2)^2 \|\boldsymbol{v}_\nu\|_2} - 2 \frac{\psi'(\|\boldsymbol{v}_\nu\|_2)^2 \langle \boldsymbol{v}_\nu, \dot{\boldsymbol{v}}_\nu \rangle \langle \boldsymbol{v}_0, \boldsymbol{v}_\nu \rangle \|\dot{\boldsymbol{v}}_\nu\|_2^2}{\psi(\|\boldsymbol{v}_\nu\|_2)^3 \|\boldsymbol{v}_\nu\|_2^2} \quad \text{(E.72)}$$

$$\left. - \frac{\psi'(\|\boldsymbol{v}_\nu\|_2) \langle \boldsymbol{v}_\nu, \dot{\boldsymbol{v}}_\nu \rangle \langle \boldsymbol{v}_0, \boldsymbol{v}_\nu \rangle \|\dot{\boldsymbol{v}}_\nu\|_2^2}{\psi(\|\boldsymbol{v}_\nu\|_2)^2 \|\boldsymbol{v}_\nu\|_2^3} \right).$$

Using the triangle inequality, Cauchy-Schwarz, and Lemma E.31, we obtain

$$\frac{\langle \boldsymbol{v}_0, \dot{\boldsymbol{v}}_\nu \rangle \psi'(\|\boldsymbol{v}_\nu\|_2) \|\dot{\boldsymbol{v}}_\nu\|_2^2}{\psi(\|\boldsymbol{v}_0\|_2)\psi(\|\boldsymbol{v}_\nu\|_2)^2 \|\boldsymbol{v}_\nu\|_2} + \frac{\langle \boldsymbol{v}_0, \boldsymbol{v}_\nu \rangle \langle \boldsymbol{v}_\nu, \dot{\boldsymbol{v}}_\nu \rangle \psi''(\|\boldsymbol{v}_\nu\|_2) \|\dot{\boldsymbol{v}}_\nu\|_2^2}{\psi(\|\boldsymbol{v}_0\|_2)\psi(\|\boldsymbol{v}_\nu\|_2)^2 \|\boldsymbol{v}_\nu\|_2^2} \leq C \|\dot{\boldsymbol{v}}_\nu\|_2^3,$$

(using also the fact that $\psi'(x) = 0$ and $\psi''(x) = 0$ whenever $x$ is sufficiently near to 0); and

$$\frac{\langle \boldsymbol{v}_0, \boldsymbol{v}_\nu \rangle \psi'(\|\boldsymbol{v}_\nu\|_2) \langle \dot{\boldsymbol{v}}_\nu, \boldsymbol{v}_\nu \rangle}{\psi(\|\boldsymbol{v}_\nu\|_2)^2 \|\boldsymbol{v}_\nu\|_2}$$

$$+ \frac{\psi'(\|\boldsymbol{v}_\nu\|_2)^2 \langle \boldsymbol{v}_\nu, \dot{\boldsymbol{v}}_\nu \rangle \langle \boldsymbol{v}_0, \boldsymbol{v}_\nu \rangle \|\dot{\boldsymbol{v}}_\nu\|_2^2}{\psi(\|\boldsymbol{v}_\nu\|_2)^3 \|\boldsymbol{v}_\nu\|_2^2} \leq C \|\dot{\boldsymbol{v}}_\nu\|_2 + C' \|\dot{\boldsymbol{v}}_\nu\|_2^3,$$

$$+ \frac{\psi'(\|\boldsymbol{v}_\nu\|_2) \langle \boldsymbol{v}_\nu, \dot{\boldsymbol{v}}_\nu \rangle \langle \boldsymbol{v}_0, \boldsymbol{v}_\nu \rangle \|\dot{\boldsymbol{v}}_\nu\|_2^2}{\psi(\|\boldsymbol{v}_\nu\|_2)^2 \|\boldsymbol{v}_\nu\|_2^3}$$

from which we conclude

$$\left| \sum_{i=1}^{n} q_i'(\nu, \boldsymbol{g}_1, \boldsymbol{g}_2) \dot{\sigma}(g_{1i} \cos \nu + g_{2i} \sin \nu) \right| \leq C \|\dot{\boldsymbol{v}}_\nu\|_2 + C' \|\dot{\boldsymbol{v}}_\nu\|_2^3$$

for some absolute constants $C, C' > 0$. By Lemma E.17, there is an event $\mathcal{E}$ of probability at least $1 - Ce^{-cn}$ on which we have $\|\dot{\boldsymbol{v}}_\nu\|_2 \leq 4$ for every $\nu$. Moreover, we have from the triangle inequality that $\|\dot{\boldsymbol{v}}_\nu\|_2 \leq \|\boldsymbol{g}_1\|_2 + \|\boldsymbol{g}_2\|_2$, which is independent of $\nu$; and in particular we have

$$\left| \sum_{i=1}^{n} q_i'(\nu, \boldsymbol{g}_1, \boldsymbol{g}_2) \dot{\sigma}(g_{1i} \cos \nu + g_{2i} \sin \nu) \right|^2 \leq \left( C(\|\boldsymbol{g}_1\|_2 + \|\boldsymbol{g}_2\|_2) + C'(\|\boldsymbol{g}_1\|_2 + \|\boldsymbol{g}_2\|_2)^3 \right)^2,$$

which is a polynomial in $\|\boldsymbol{g}_1\|_2$ and $\|\boldsymbol{g}_2\|_2$ by the binomial theorem. Thus, applying independence, Lemma G.10, Lemma G.11 yields that there is an absolute constant $C'' > 0$ such that

$$\mathop{\mathbb{E}}_{\boldsymbol{g}_1, \boldsymbol{g}_2} \left[ \left( C(\|\boldsymbol{g}_1\|_2 + \|\boldsymbol{g}_2\|_2) + C'(\|\boldsymbol{g}_1\|_2 + \|\boldsymbol{g}_2\|_2)^3 \right)^2 \right] \leq C''.$$

Therefore, as in the framework section of the proof of Lemma E.13, we can use the inequality

$$\left| \mathop{\mathbb{E}}_{\boldsymbol{g}_2} \left[ \sum_{i=1}^{n} q_i'(\nu, \boldsymbol{g}_1, \boldsymbol{g}_2) \dot{\sigma}(g_{1i} \cos \nu + g_{2i} \sin \nu) \right] \right| \leq \mathop{\mathbb{E}}_{\boldsymbol{g}_2} \left[ \left| \sum_{i=1}^{n} q_i'(\nu, \boldsymbol{g}_1, \boldsymbol{g}_2) \dot{\sigma}(g_{1i} \cos \nu + g_{2i} \sin \nu) \right| \right],$$

$$\text{(E.73)}$$

together with the partition

$$\mathop{\mathbb{E}}_{\boldsymbol{g}_2} \left[ \left| \sum_{i=1}^{n} q_i'(\nu, \boldsymbol{g}_1, \boldsymbol{g}_2) \dot{\sigma}(g_{1i} \cos \nu + g_{2i} \sin \nu) \right| \right]$$

$$\leq C' + \mathop{\mathbb{E}}_{\boldsymbol{g}_2} \left[ \mathbb{1}_{(\mathcal{E}')^c} \left| \sum_{i=1}^{n} q_i'(\nu, \boldsymbol{g}_1, \boldsymbol{g}_2) \dot{\sigma}(g_{1i} \cos \nu + g_{2i} \sin \nu) \right| \right], \quad \text{(E.74)}$$

and this last expression can be used to obtain a $\boldsymbol{g}_1$ event of not much smaller probability $1 - Ce^{-cn}$ on which the LHS of (E.74), and hence the LHS of (E.73), is controlled by an absolute constant

uniformly in $\nu$ (in particular, using Markov's inequality as in the framework section of the proof of Lemma E.13). Consequently, one more application of the triangle inequality gives that

$$\mathbb{P}\left[\left|\mathbb{E}_{\boldsymbol{g}_2}[\Xi_5(\nu, \boldsymbol{g}_1, \boldsymbol{g}_2)] - \mathbb{E}_{\boldsymbol{g}_1, \boldsymbol{g}_2}[\Xi_5(\nu, \boldsymbol{g}_1, \boldsymbol{g}_2)]\right| \text{ is } C + C'n^{1+\delta}\text{-Lipschitz}\right] \geq 1 - C''e^{-cn} - C'''n^{-\delta}$$

as long as $\delta \geq \frac{1}{2}$. $\qquad\square$

**Lemma E.46.** *In the notation of Lemma E.13, if $d \geq 1$, there are absolute constants $c, C, C', C'' > 0$ and absolute constants $K, K' > 0$ such that if $n \geq Kd\log n$ and $d \geq K'$, there is an event with probability at least $1 - Ce^{-cn} - C'n^{-d}$ on which one has*

$$\forall \nu \in [0, \pi], \ |\Xi_6(\nu, \boldsymbol{g}_1, \boldsymbol{g}_2) - \mathbb{E}[\Xi_6(\nu, \boldsymbol{g}_1, \boldsymbol{g}_2)]| \leq C''\sqrt{\frac{d\log n}{n}}.$$

*Proof.* The argument is extremely similar to Lemma E.43, since both terms have small expectations and deviations essentially determinable by the same mean-zero random variable.

We are going to control the expectation first, showing that it is small; then prove that $|\Xi_6|$ is small uniformly in $\nu$. Let $\mathcal{E}$ denote the event $\mathcal{E}_{0.5,0}$ in Lemma E.16; then by that lemma, $\mathcal{E}$ has probability at least $1 - Ce^{-cn}$ as long as $n \geq C'$, where $c, C, C' > 0$ are absolute constants, and for $(\boldsymbol{g}_1, \boldsymbol{g}_2) \in \mathcal{E}$, one has for all $\nu \in [0, \pi]$

$$\Xi_6(\nu, \boldsymbol{g}_1, \boldsymbol{g}_2) = 3\frac{\langle \boldsymbol{v}_0, \boldsymbol{v}_\nu \rangle \langle \boldsymbol{v}_\nu, \dot{\boldsymbol{v}}_\nu \rangle^2}{\|\boldsymbol{v}_0\|_2 \|\boldsymbol{v}_\nu\|_2^5}.$$

Thus, if we write

$$\widetilde{\Xi}_6(\nu, \boldsymbol{g}_1, \boldsymbol{g}_2) = 3\mathbb{1}_{\mathcal{E}}(\boldsymbol{g}_1, \boldsymbol{g}_2)\frac{\langle \boldsymbol{v}_0, \boldsymbol{v}_\nu \rangle \langle \boldsymbol{v}_\nu, \dot{\boldsymbol{v}}_\nu \rangle^2}{\|\boldsymbol{v}_0\|_2 \|\boldsymbol{v}_\nu\|_2^5},$$

we have $\widetilde{\Xi}_6 = \Xi_6$ for all $\nu$ whenever $(\boldsymbol{g}_1, \boldsymbol{g}_2) \in \mathcal{E}$, so that for any $\nu$

$$|\mathbb{E}[\Xi_6(\nu, \boldsymbol{g}_1, \boldsymbol{g}_2)]| = \left|\mathbb{E}[\widetilde{\Xi}_6(\nu, \boldsymbol{g}_1, \boldsymbol{g}_2)] + \mathbb{E}[\mathbb{1}_{\mathcal{E}^c}\Xi_6(\nu, \boldsymbol{g}_1, \boldsymbol{g}_2)]\right|$$

$$\leq |\mathbb{E}[\widetilde{\Xi}_6(\nu, \boldsymbol{g}_1, \boldsymbol{g}_2)]| + Ce^{-cn},$$

where the second line uses the triangle inequality and the Schwarz inequality and Lemma E.37 together with the Lyapunov inequality. We proceed with analyzing the expectation of $\widetilde{\Xi}_6$. Using the Schwarz inequality gives

$$\left|\mathbb{E}\left[\widetilde{\Xi}_6(\nu, \boldsymbol{g}_1, \boldsymbol{g}_2)\right]\right| \leq 3\mathbb{E}\left[\langle \boldsymbol{v}_0, \boldsymbol{v}_\nu \rangle^2 \langle \boldsymbol{v}_\nu, \dot{\boldsymbol{v}}_\nu \rangle^4\right]^{1/2}\mathbb{E}\left[\frac{\mathbb{1}_{\mathcal{E}}}{\|\boldsymbol{v}_0\|_2^2\|\boldsymbol{v}_\nu\|_2^{10}}\right]^{1/2}$$

$$\leq 192\mathbb{E}\left[\langle \boldsymbol{v}_0, \boldsymbol{v}_\nu \rangle^2 \langle \boldsymbol{v}_\nu, \dot{\boldsymbol{v}}_\nu \rangle^4\right]^{1/2},$$

and the checks at and around (E.36) in the proof of Lemma E.15 show that we can apply Lemma E.30 to obtain

$$\left|\mathbb{E}\left[\langle \boldsymbol{v}_0, \boldsymbol{v}_\nu \rangle^2 \langle \boldsymbol{v}_\nu, \dot{\boldsymbol{v}}_\nu \rangle^4\right]\right.$$

$$\left. - n^6\mathbb{E}[\sigma(g_{11})\sigma(g_{11}\cos\nu + g_{21}\sin\nu)]^2\mathbb{E}[\sigma(g_{11}\cos\nu + g_{21}\sin\nu)(g_{21}\cos\nu - g_{11}\sin\nu)]^4\right| \leq \frac{C}{n}.$$

But we have using rotational invariance that $\mathbb{E}[\sigma(g_{11}\cos\nu + g_{21}\sin\nu)(g_{21}\cos\nu - g_{11}\sin\nu)] = 0$, which implies

$$\left|\mathbb{E}\left[\langle \boldsymbol{v}_0, \boldsymbol{v}_\nu \rangle^2 \langle \boldsymbol{v}_\nu, \dot{\boldsymbol{v}}_\nu \rangle^4\right]\right| \leq C/n,$$

from which we conclude for all $\nu$

$$|\mathbb{E}[\Xi_6(\nu, \boldsymbol{g}_1, \boldsymbol{g}_2)]| \leq C/\sqrt{n}.$$

Next, we control the deviations of $\Xi_6$ with high probability. By Lemma E.17, there is an event $\mathcal{E}_a$ with probability at least $1 - e^{-cn}$ on which $\|\dot{\boldsymbol{v}}_\nu\|_2 \leq 4$ for every $\nu \in [0, \pi]$. Therefore on the

event $\mathcal{E}_b = \mathcal{E} \cap \mathcal{E}_a$, which has probability at least $1 - Ce^{-cn}$ by a union bound, we have using Cauchy-Schwarz that for every $\nu$

$$|\Xi_6(\nu, \boldsymbol{g}_1, \boldsymbol{g}_2)| \le 6144|\langle \boldsymbol{v}_\nu, \dot{\boldsymbol{v}}_\nu \rangle|.$$

Using the high probability deviations bound established in (E.64), it follows that if $n$ is large enough then with probability at least $1 - Ce^{-cn} - C'n^{-2d+1/2}$ we have

$$\forall \nu \in [0, \pi], \ |\Xi_6(\nu, \boldsymbol{g}_1, \boldsymbol{g}_2)| \le C'' \sqrt{\frac{d \log n}{n}}.$$

As long as $d \ge \frac{1}{2}$, we have that this probability is at least $1 - Ce^{-cn} - C'n^{-d}$, and so the triangle inequality yields finally that with probability at least $1 - Ce^{-cn} - C'n^{-d}$

$$\forall \nu \in [0, \pi], \ |\Xi_6(\nu, \boldsymbol{g}_1, \boldsymbol{g}_2) - \mathbb{E}[\Xi_6(\nu, \boldsymbol{g}_1, \boldsymbol{g}_2)]| \le C'' \sqrt{\frac{d \log n}{n}}.$$

$\square$

**Lemma E.47.** *Consider the function*

$$g(\nu) = -(\pi^2 - [(\pi-\nu)\cos\nu + \sin\nu]^2)[(\pi-\nu)\cos\nu - \sin\nu] + (\pi-\nu)^2[(\pi-\nu)\cos\nu + \sin\nu]\sin^2\nu,$$

*which is the negated numerator of $\ddot{\varphi}$. Then if $0 \le \nu \le \pi/2$, one has a bound*

$$\frac{2\pi^2}{3}\nu^3 - \frac{83}{24}\nu^4 \le g(\nu),$$

*and the lower bound is positive if $0 < \nu \le \pi/2$.*

*Proof.* To see that the lower bound is positive under the stated condition, write

$$\frac{2\pi^2}{3}\nu^3 - \frac{83}{24}\nu^4 = \nu^3\left(\frac{2\pi^2}{3} - \frac{83}{24}\nu\right);$$

the quantity in parentheses is positive in a neighborhood of zero by continuity, and in fact one calculates for its unique zero $\nu_0 = 48\pi^2/249$, and one verifies numerically that $48\pi^2/249 > 1.9 > \pi/2$. We conclude that the bound is positive for $0 < \nu < 1.9$ by continuity.

To establish the bound, we employ Taylor expansion of the numerator, which is a smooth function on $(0, \pi)$ with continuous derivatives of all orders on $[0, \pi]$, in a neighborhood of zero. In our development in the proof of Lemma E.5, we showed that the analytic function $-g(\nu) = -(2\pi^2/3)\nu^3 + O(\nu^4)$ near zero, so Taylor's theorem with Lagrange remainder implies

$$\frac{2\pi^3}{3}\nu^3 + \frac{\nu^4}{24}\inf_{\nu \in [0, \pi/2]} g^{(4)}(\nu) \le g(\nu),$$

and so it suffices to get suitable bounds on the fourth derivative of $g$. We will develop the bounds rather tediously. Start by distributing in $g$ to write

$$g(\nu) = \nu^3 \underbrace{(-\cos\nu)}_{g_3(\nu)} + \nu^2 \underbrace{(3\pi\cos\nu + \sin\nu)}_{g_2(\nu)} + \nu \underbrace{(\cos\nu - 2\pi^2\cos\nu - 2\pi\sin\nu - \cos^3\nu)}_{g_1(\nu)}$$
$$+ \underbrace{(\pi\cos^3\nu + 2\pi^2\sin\nu - \sin^3\nu - \pi\cos\nu)}_{g_0(\nu)}.$$

Using the Leibniz rule, we have for the fourth derivative

$$g^{(4)}(\nu) = \nu^3\left(g_3^{(4)}(\nu)\right) + \nu^2\left(g_2^{(4)}(\nu) + 12g_3^{(3)}(\nu)\right) + \nu\left(g_1^{(4)}(\nu) + 8g_2^{(3)}(\nu) + 36g_3^{(2)}(\nu)\right)$$
$$+ \left(g_0^{(4)}(\nu) + 4g_1^{(3)}(\nu) + 12g_2^{(2)}(\nu) + 24g_3^{(1)}(\nu)\right).$$

To calculate these derivatives, we just need to differentiate $\sin$, $\cos$, and their third powers. Write $c(\nu) = \cos^3(\nu)$ and $s(\nu) = \sin^3(\nu)$; using the elementary calculations

$$
\begin{aligned}
c^{(1)}(\nu) &= 3s(\nu) - 3\sin\nu, \quad c^{(2)}(\nu) = 6\cos\nu - 9c(\nu), \\
c^{(3)}(\nu) &= 21\sin\nu - 27s(\nu), \quad c^{(4)}(\nu) = 60\cos\nu + 81c(\nu); \\
s^{(1)}(\nu) &= 3\cos\nu - 3c(\nu), \quad c^{(2)}(\nu) = 6\sin\nu - 9s(\nu), \\
c^{(3)}(\nu) &= 27c(\nu) - 21\cos\nu, \quad c^{(4)}(\nu) = 60\sin\nu + 81s(\nu),
\end{aligned}
\tag{E.75}
$$

one can calculate the results

$$
g_3^{(4)}(\nu) = -\cos\nu, \quad g_2^{(4)}(\nu) = 3\pi\cos\nu + \sin\nu,
$$
$$
g_1^{(4)}(\nu) = (61 - 2\pi^2)\cos\nu - 2\pi\sin\nu - 81\cos^3\nu,
$$
$$
g_0^{(4)}(\nu) = (2\pi^2 - 60)\sin\nu + 50\pi\cos\nu + 81\pi\cos^3\nu - 81\sin^3\nu;
$$

and

$$
g_3^{(3)}(\nu) = -\sin\nu, \quad g_2^{(3)}(\nu) = 3\pi\sin\nu - \cos\nu,
$$
$$
g_1^{(3)}(\nu) = (7 - 2\pi^2)\sin\nu + 2\pi\cos\nu - 27\cos^2\nu\sin\nu;
$$

and

$$
g_3^{(2)}(\nu) = \cos\nu \quad g_2^{(2)}(\nu) = -3\pi\cos\nu - \sin\nu;
$$

and finally

$$
g_3^{(1)}(\nu) = \sin\nu.
$$

Plugging back into (E.75) and canceling, we get

$$
g^{(4)}(\nu) = \nu^3 \underbrace{(-\cos\nu)}_{h_3(\nu)} + \nu^2 \underbrace{(3\pi\cos\nu - 11\sin\nu)}_{h_2(\nu)} + \nu \underbrace{(22\pi\sin\nu + (89 - 2\pi^2)\cos\nu - 81\cos^3\nu)}_{h_1(\nu)}
$$
$$
+ \underbrace{(27\sin^3\nu + 81\pi\cos^3\nu + 31\pi\cos\nu - (6\pi^2 + 128)\sin\nu)}_{h_0(\nu)}.
$$

Since $\nu > 0$, we can leverage lower bounds on each $h_i$ term. We have trivially $|h_3| \leq 1$, so that $|\nu^3 h_3(\nu)| \leq \pi^3/8$. We will study $\nu h_1(\nu) + h_0(\nu)$ together to get a better bound. We have

$$
\nu h_1(\nu) + h_0(\nu) = \left(22\pi\nu - (6\pi^2 + 128)\right)\sin\nu + 27\sin^3\nu + \left((89 - 2\pi^2)\nu + 31\pi\right)\cos\nu
$$
$$
+ (81\pi - 81\nu)\cos^3\nu
$$
$$
\geq \underbrace{\left(22\pi\nu - (6\pi^2 + 128)\right)\sin\nu + 27\sin^3\nu + \left((89 - 2\pi^2)\nu + 31\pi\right)\cos\nu}_{q(\nu)},
$$

$$\tag{E.76}$$

using $\nu \leq \pi/2$ and $\cos \geq 0$ on this domain. We will show that the RHS of the final inequality, denoted $q$, is a decreasing function of $\nu$, and is therefore lower bounded by its value at $\nu = \pi/2$ on our interval of interest. We calculate

$$
q'(\nu) = 9\pi\sin\nu + (42 - 8\pi^2)\cos\nu + 22\pi\nu\cos\nu - (89 - 2\pi^2)\nu\sin\nu - 81\cos^3\nu.
$$

Reordering terms, we can write

$$
q'(\nu) = -81\cos^3\nu + \left(\underbrace{9\pi}_{C_1} - \underbrace{(89 - 2\pi^2)}_{C_2}\nu\right)\sin\nu - \left(\underbrace{(8\pi^2 - 42)}_{C_3} - \underbrace{22\pi}_{C_4}\nu\right)\cos\nu. \tag{E.77}
$$

We can estimate numerically

$$
69 \leq C_2 \leq 70; \quad 69 \leq C_4 \leq 70; \quad C_2 > C_4,
$$

which shows that $C_1, C_2, C_3, C_4 > 0$ and both of the linear prefactors are decreasing functions of $\nu$. We have on all of $(0, \pi/2)$ by concavity of $\sin$

$$
(C_1 - C_2\nu)\sin\nu \leq \nu\left(C_1 - \frac{2C_2}{\pi}\nu\right),
$$

using in particular $\sin \nu \leq \nu$ and $\sin \nu \geq (2/\pi)\nu$. Using similarly concavity of $\cos$ on this domain, in particular the inequalities $\cos \nu \leq \pi/2 - \nu$ and $\cos \nu \geq 1 - (2/\pi)\nu$, we have

$$-(C_3 - C_4\nu)\cos \nu \leq -\left(C_4\nu^2 - \left(\frac{2C_3}{\pi} + \frac{C_4\pi}{2}\right)\nu + C_3\right).$$

In total, we have a bound

$$q'(\nu) \leq -81\cos^3 \nu - \left(\frac{2C_2}{\pi} + C_4\right)\nu^2 + \left(\frac{2C_3}{\pi} + \frac{\pi C_4}{2} + C_1\right)\nu - C_3.$$

We calculate the maximizer of the concave quadratic function of $\nu$ in the previous bound via differentiation; plugging in, we get

$$q'(\nu) \leq -81\cos^3 \nu + \frac{\left(\frac{2C_3}{\pi} + \frac{\pi C_4}{2} + C_1\right)^2}{4\left(\frac{2C_2}{\pi} + C_4\right)} - C_3.$$

A numerical estimate gives

$$\frac{\left(\frac{2C_3}{\pi} + \frac{\pi C_4}{2} + C_1\right)^2}{4\left(\frac{2C_2}{\pi} + C_4\right)} - C_3 \leq 20,$$

and using that $-\cos^3$ is strictly decreasing for $\nu < \pi$, we can therefore guarantee $q' \leq 0$ as long as $\nu \leq \cos^{-1}\sqrt[3]{20/81}$. Writing $c = \cos^{-1}\sqrt[3]{20/81}$, we estimate numerically $0.90 \geq c \geq 0.89$, so that this bound is nonvacuous. For $\nu \geq c$, we apply again concavity of $\cos$ to develop the lower bound

$$\cos \nu \geq \left(\frac{\pi/2 - \nu}{\pi/2 - c}\right)\cos c, \quad \nu \in [c, \pi/2].$$

Using this to estimate the $-\cos^3$ term in our upper bound for $q'$, we obtain a bound

$$q'(\nu) \leq -20\left(\frac{\pi/2 - \nu}{\pi/2 - c}\right)^3 - \left(\frac{2C_2}{\pi} + C_4\right)\nu^2 + \left(\frac{2C_3}{\pi} + \frac{\pi C_4}{2} + C_1\right)\nu - C_3, \quad c \leq \nu \leq \pi/2.$$

We define $D = 20/(\pi/2 - c)^3$, $A = 2C_2/\pi + C_4$, $B = 2C_3/\pi + \pi C_4/2 + C_1$, and $C = C_3$, so that the RHS can be written as $-D(\pi/2 - \nu)^3 - A\nu^2 + B\nu - C$. Differentiating once and equating to zero results in the quadratic equation

$$3D\left(\nu^2 - \underbrace{\left(\frac{2A}{3D} + \pi\right)}_{M}\nu + \underbrace{\left(\frac{B}{3D} + \pi^2/4\right)}_{N}\right) = 0,$$

which has roots $M/2 \pm \frac{1}{2}\sqrt{M^2 - 4N}$. Numerically estimating the constants, we get that the two roots lie in $[0.99, 1]$ and $[3.3, 3.4]$, so that we need only consider the smaller root. Differentiating once more to determine the class of the critical point, we find for the second derivative at $M/2 - \frac{1}{2}\sqrt{M^2 - 4N}$

$$-3D\sqrt{M^2 - 4N} < 0,$$

so that $M/2 - \frac{1}{2}\sqrt{M^2 - 4N}$ is a maximizer for our cubic bound, and the bound is increasing for arguments less than this point and decreasing for arguments greater than it; we can conclude that the zero in $[3.3, 3.4]$ is a minimizer, so that our bound can be ascertained negative by checking its value at $M/2 - \frac{1}{2}\sqrt{M^2 - 4N}$. We find using a numerical estimate

$$-20\left(\frac{\pi/2 - (M/2 - \frac{1}{2}\sqrt{M^2 - 4N})}{\pi/2 - c}\right)^3$$
$$-\left(\frac{2C_2}{\pi} + C_4\right)(M/2 - \frac{1}{2}\sqrt{M^2 - 4N})^2$$
$$+\left(\frac{2C_3}{\pi} + \frac{\pi C_4}{2} + C_1\right)(M/2 - \frac{1}{2}\sqrt{M^2 - 4N}) - C_3 \leq -1.7 < 0,$$

which proves that $q' \leq 0$ on $[c, \pi/2]$. This shows that our lower bound on $\nu h_1(\nu) + h_0(\nu)$ in (E.76) is nonincreasing on $[0, \pi/2]$, so that we can assert

$$
\begin{aligned}
\nu h_1(\nu) + h_0(\nu) &\geq \big(22\pi(\pi/2) - (6\pi^2 + 128)\big) \sin(\pi/2) + 27\sin^3(\pi/2) \\
&\quad + \big((89 - 2\pi^2)(\pi/2) + 31\pi\big)\cos(\pi/2) \\
&= 5\pi^2 - 101.
\end{aligned}
$$

It remains to bound $\nu^2 h_2(\nu) = \nu^2(3\pi\cos\nu - 11\sin\nu)$. On $[0, \pi/2]$, cos is decreasing and sin is increasing, so $3\pi\cos\nu - 11\sin\nu$ is decreasing here; it is positive at $\nu = 0$ and negative at $\nu = \pi/2$, so that by continuity it has a unique zero in $(0, \pi/2)$. Denote this zero as $\nu_0$; then using that $\nu^2 \geq 0$ with no zeros in the interior, we can write

$$
\inf_{0 \leq \nu \leq \nu_0} \nu^2 h_2(\nu) \geq 0,
$$

and

$$
\inf_{\nu_0 \leq \nu \leq \pi/2} \nu^2 h_2(\nu) \geq \left( \sup_{\nu_0 \leq \nu \leq \pi/2} \nu^2 \right) \left( \inf_{\nu_0 \leq \nu \leq \pi/2} h_2(\nu) \right)
$$

$$
\geq (\pi/2)^2 (3\pi\cos(\pi/2) - 11\sin(\pi/2)) = -\frac{11\pi^2}{4},
$$

which gives the bound $\nu^2 h_2(\nu) \geq -11\pi^2/4$ on $[0, \pi/2]$. Putting it all together, we have

$$
g^{(4)}(\nu) \geq -\frac{11\pi^2}{4} + 5\pi^2 - 101 - \pi^3/8 \geq -83,
$$

where the last inequality follows from a numerical estimate of the constants. $\qquad\square$

**Lemma E.48** (Uniformization). *Let $(\Omega, \mathcal{F}, \mathbb{P})$ be a complete probability space. For some $t \in \mathbb{R}$, $\delta_t \geq 0$, $S \subset \mathbb{R}^d$, and event $\mathcal{E} \in \mathcal{F}$, suppose that $f : S \times \Omega \to \mathbb{R}$ is second-argument measurable and satisfies*

1. *For all $\boldsymbol{x} \in S$, $\mathbb{P}[f(\boldsymbol{x}, \cdot) \leq t] \geq 1 - \delta_t$;*

2. *For all $\boldsymbol{g} \in \mathcal{E}$, $f(\cdot, \boldsymbol{g})$ is L-Lipschitz;*

3. *There is $M > 0$ such that $\sup_{\boldsymbol{x} \in S} \|\boldsymbol{x}\|_2 \leq M$.*

*Then $\boldsymbol{g} \mapsto \sup_{\boldsymbol{x} \in S} f(\boldsymbol{x}, \boldsymbol{g})$ is measurable, and for every $\varepsilon > 0$, one has*

$$
\mathbb{P}\left[ \sup_{x \in S} f(\boldsymbol{x}, \cdot) \leq t + L\varepsilon \right] \geq 1 - \delta_t \left(1 + \frac{2M}{\varepsilon}\right)^d - \mathbb{P}[\mathcal{E}]. \tag{E.78}
$$

*Proof.* Because $S$ is a subset of the separable metric space $(\mathbb{R}^d, \|\cdot\|_2)$ and all sample trajectories $f(\cdot, \boldsymbol{g})$ are assumed (Lipschitz) continuous, the supremum in the definition of $\boldsymbol{g} \mapsto \sup_{\boldsymbol{x} \in S} f(\boldsymbol{x}, \boldsymbol{g})$ can be taken on a countable subset of $S$, and the resulting function of $\boldsymbol{g}$ is measurable (e.g., (Ledoux & Talagrand, 1991, §2.2 p. 45)). By (Vershynin, 2018, Proposition 4.2.12) and boundedness of $S$, for every $\varepsilon > 0$ there exists an $\varepsilon$-net of $S$ having cardinality at most $(1 + 2M/\varepsilon)^d$; denote these nets as $N_\varepsilon$. Since each $N_\varepsilon$ is finite, we may also define for each $\boldsymbol{x} \in S$ a point $\boldsymbol{x}_\varepsilon$ such that $\|\boldsymbol{x} - \boldsymbol{x}_\varepsilon\|_2 \leq \varepsilon$; then for every $\boldsymbol{g} \in \mathcal{E}$, we have $|f(\boldsymbol{x}, \boldsymbol{g}) - f(\boldsymbol{x}_\varepsilon, \boldsymbol{g})| \leq L\varepsilon$. We define a collection of events $\mathcal{E}_\varepsilon$ by

$$
\mathcal{E}_\varepsilon = \{\boldsymbol{g} \in \Omega \mid \forall \boldsymbol{x} \in N_\varepsilon, f(\boldsymbol{x}, \boldsymbol{g}) \leq t\}. \tag{E.79}
$$

The triangle inequality then implies that if $\boldsymbol{g} \in \mathcal{E}_\varepsilon \cap \mathcal{E}$, then for all $\boldsymbol{x} \in S$, one has $f(\boldsymbol{x}, \boldsymbol{g}) \leq t + L\varepsilon$. Consequently, several union bounds yield

$$
\begin{aligned}
\mathbb{P}\left[ \sup_{\boldsymbol{x} \in S} f(\boldsymbol{x}, \cdot) > t + L\varepsilon \right] &\leq \mathbb{P}\left[ \sup_{\boldsymbol{x} \in N_\varepsilon} f(\boldsymbol{x}, \boldsymbol{g}) \leq t \right] + \mathbb{P}[\mathcal{E}] \\
&\leq \delta_t \left(1 + \frac{2M}{\varepsilon}\right)^d + \mathbb{P}[\mathcal{E}],
\end{aligned} \tag{E.80}
$$

as claimed. $\qquad\square$

### E.4 DEFERRED PROOFS

*Proof of Lemma E.5.* The function $\cos^{-1}$ is $C^{\infty}$ on $(-1, 1)$, and because $f(\nu) := \cos\varphi(\nu)$ is smooth and satisfies $f'(\nu) = (\pi^{-1}\nu - 1)\sin\nu < 0$ if $\nu < \pi$ with $f(0) = 1$ and $f(\pi) = 0$, we see that $\varphi$ is $C^{\infty}$ on $(0, \pi)$ by the chain rule. This also shows $\varphi(0) = \cos^{-1}(1) = 0$ and $\varphi(\pi) = \cos^{-1}(0) = \pi/2$. Direct calculation gives

$$\dot{\varphi}(\nu) = \sqrt{\frac{(\pi - \nu)^2 \sin^2\nu}{\pi^2 - ((\pi - \nu)\cos\nu + \sin\nu)^2}} \tag{E.81}$$

and

$$\begin{aligned}
\ddot{\varphi}(\nu) = {} & \frac{(\pi^2 - [(\pi - \nu)\cos\nu + \sin\nu]^2)[(\pi - \nu)\cos\nu - \sin\nu]}{(\pi^2 - [(\pi - \nu)\cos\nu + \sin\nu]^2)^{3/2}} \\
& - \frac{(\pi - \nu)^2[(\pi - \nu)\cos\nu + \sin\nu]\sin^2\nu}{(\pi^2 - [(\pi - \nu)\cos\nu + \sin\nu]^2)^{3/2}}
\end{aligned} \tag{E.82}$$

Calculating endpoint limits using these expressions will suffice to show the derivatives are continuous on $[0, \pi]$ and give the claimed values there. We have

$$\begin{aligned}
\lim_{\nu\searrow 0}\left(\dot{\varphi}(\nu)\right)^2 &= \lim_{\nu\searrow 0}\frac{(\pi - \nu)^2 \sin^2\nu}{\pi^2 - ((\pi - \nu)\cos\nu + \sin\nu)^2} \\
&= \lim_{\nu\searrow 0}\frac{2(\pi - \nu)\sin\nu[(\pi - \nu)\cos\nu - \sin\nu]}{(-2)[(\pi - \nu)\cos\nu + \sin\nu][\cos\nu - (\pi - \nu)\sin\nu - \cos\nu]} \\
&= \lim_{\nu\searrow 0}\frac{(\pi - \nu)\cos\nu - \sin\nu}{(\pi - \nu)\cos\nu + \sin\nu} = 1,
\end{aligned}$$

by L'Hôpital's rule, whereas a direct evaluation gives

$$\lim_{\nu\searrow 0}\left(\dot{\varphi}(\nu)\right)^2 = \frac{0}{\pi^2} = 0.$$

Continuity of the square root function gives the claimed results for $\dot{\varphi}$. Again by direct calculation, we find

$$\lim_{\nu\searrow 0}\left(\ddot{\varphi}(\nu)\right)^2 = \frac{0}{\pi^3} = 0.$$

Since $\dot{\varphi}^2$ is meromorphic in a neighborhood of $0$ with, as we have shown, a removable singularity at $0$, it is actually analytic, and we can calculate further derivatives at $0$ by expanding it locally at $0$. We use the expansions $\sin\nu = \nu - \nu^3/6 + O(\nu^5)$ and $\cos\nu = 1 - \nu^2/2 + \nu^4/24 + O(\nu^6)$ near $0$ to calculate

$$\left(1 - \frac{\nu}{\pi}\right)^2 \sin^2\nu = \nu^2\left(1 - \frac{2}{\pi}\nu - \frac{\pi^2 - 3}{3\pi^2}\nu^2 + O(\nu^3)\right)$$

and

$$1 - \left(\left(1 - \frac{\nu}{\pi}\right)\cos\nu + \frac{\sin\nu}{\pi}\right)^2 = \nu^2\left(1 - \frac{2}{3\pi}\nu - \frac{1}{3}\nu^2 + O(\nu^3)\right),$$

from which it follows

$$(\dot{\varphi}(\nu))^2 = \left(1 - \frac{2}{\pi}\nu - \frac{\pi^2 - 3}{3\pi^2}\nu^2 + O(\nu^3)\right)\left(1 - \frac{2}{3\pi}\nu - \frac{1}{3}\nu^2 + O(\nu^3)\right)^{-1}.$$

By the geometric series, we then obtain

$$(\dot{\varphi}(\nu))^2 = 1 - \frac{4}{3\pi}\nu + \frac{1}{9\pi^2}\nu^2 + O(\nu^3).$$

Taking the square root of this expression and applying the binomial series, we thus have

$$\dot{\varphi}(\nu) = 1 - \frac{2}{3\pi}\nu - \frac{1}{6\pi^2}\nu^2 + O(\nu^3),$$

from which we read off

$$\lim_{\nu\searrow 0}\ddot{\varphi}(\nu) = -\frac{2}{3\pi}; \qquad \lim_{\nu\searrow 0}\dddot{\varphi}(\nu) = -\frac{1}{3\pi^2}.$$

It is clear from the analytical expression for $\dot{\varphi}$ and the mean value theorem that $\varphi$ is strictly increasing on $[0, \pi]$, since $(\pi - \nu) \sin \nu > 0$ if $0 < \nu < \pi$. To prove strict concavity for $\nu \in (0, \pi)$, we start by simplifying notation. Consider the function $\varphi_r(\nu) = \varphi(\pi - \nu)$, which satisfies by the chain rule $\ddot{\varphi}_r(\nu) = \ddot{\varphi}(\pi - \nu)$. Because $\varphi_r$ is strictly concave if and only if $\varphi$ is strictly concave, it suffices to prove that $\ddot{\varphi}(\pi - \nu) < 0$. We note

$$\ddot{\varphi}(\pi - \nu) < 0 \iff (\pi^2 - [\nu \cos \nu - \sin \nu]^2)(-\sin \nu - \nu \cos \nu) < \nu^2 \sin^2 \nu (\sin \nu - \nu \cos \nu).$$

Multiplying both sides of the latter inequality by $\sin \nu - \nu \cos \nu$, dividing through by $(\nu \cos \nu - \sin \nu)^2$ (which is positive on $(0, \pi)$, since it equals $\cos^2 \varphi$ composed with a reversal about $\pi$), and distributing and moving terms to the RHS gives the equivalent condition

$$\pi^2 \frac{\nu^2 \cos^2 \nu - \sin^2 \nu}{(\nu \cos \nu - \sin \nu)^2} < \nu^2 - \sin^2 \nu,$$

and canceling once more gives equivalently

$$\frac{\nu \cos \nu + \sin \nu}{\nu \cos \nu - \sin \nu} < \frac{\nu^2 - \sin^2 \nu}{\pi^2}. \tag{E.83}$$

Using $\nu \cos \nu - \sin \nu < 0$, which follows from its derivative $-\nu \sin \nu$ being negative on $(0, \pi)$, and writing $g(\nu) = \pi^{-2}(\nu^2 - \sin^2 \nu)$, we have equivalently $\nu \cos \nu + \sin \nu > g(\nu)(\nu \cos \nu - \sin \nu)$, and rearranging gives the inequality

$$(1 - g(\nu))\nu \cos \nu + g(\nu) \sin \nu > -\sin \nu. \tag{E.84}$$

Strict concavity of $\sin$ on $(0, \pi)$ gives $\sin \nu < \nu$, and $0 < g(\nu) < 1$ follows after squaring; so the LHS is a convex combination of $\nu \cos \nu$ and $\sin \nu$, which in particular satisfies $|(1 - g(\nu))\nu \cos \nu + g(\nu) \sin \nu| \leq \max\{|\sin \nu|, |\nu \cos \nu|\}$. As argued before, we have $\sin \nu - \nu \cos \nu > 0$ if $\nu \in (0, \pi)$; moreover, because $\nu > 0$ we have $\nu \cos \nu > 0$ if $\nu \in (0, \pi/2)$ and $\nu \cos \nu < 0$ if $\nu \in (\pi/2, \pi)$. We can numerically determine $\sin(5\pi/8) + (5\pi/8) \cos(5\pi/8) > 0$, and given that $5\pi/8 \geq 1.95 > \pi/2$, it follows

$$|(1 - g(\nu))\nu \cos \nu + g(\nu) \sin \nu| < |\sin \nu|, \quad 0 < \nu \leq 1.95,$$

which implies (E.84) when $0 < \nu \leq 1.95$. Recalling that we are arguing for $\varphi_r$ in this setting, we translate our results back to $\varphi$ and conclude that $\varphi(\nu) < 0$ if $\pi - 1.95 \leq \nu < \pi$. To address the case where $0 < \nu < \pi - 1.95$, we employ Lemma E.47; it allows us to conclude $\ddot{\varphi} < 0$ provided $0 < \nu \leq \pi/2$, and a numerical estimate gives that $\pi - 1.95 < \pi/2$, so that we have $\ddot{\varphi} < 0$ for all $0 < \nu < \pi$. Taking limits in $\varphi$ gives concavity at the endpoints $\{0, \pi\}$ as well.

To bound $\ddot{\varphi}$ away from zero on $[0, \pi/2]$, we apply Lemma E.47 to assert

$$\ddot{\varphi}(\nu) \leq \frac{-\frac{2}{3\pi} \nu^3 + \frac{83}{24\pi^3} \nu^4}{(1 - \cos^2 \varphi(\nu))^{3/2}}, \quad 0 < \nu \leq \pi/2.$$

The numerator in the last expression is nonpositive if $0 \leq \nu \leq \pi/2$, and using the lower bound in Lemma E.14 on $[0, \pi/2]$, we have

$$\frac{1}{1 - \cos^2 \varphi(\nu)} \geq \frac{1}{1 - \max^2\{1 - \frac{1}{2}\nu^2, 0\}}, \quad \nu > 0.$$

From nonpositivity of the numerator, it follows

$$\ddot{\varphi}(\nu) \leq \frac{-\frac{2}{3\pi} \nu^3 + \frac{83}{24\pi^3} \nu^4}{\left(1 - \max^2\{1 - \frac{1}{2}\nu^2, 0\}\right)^{3/2}}, \quad 0 < \nu \leq \pi/2. \tag{E.85}$$

We have $1 - \frac{1}{2}\nu^2 \geq 0$ as long as $0 \leq \nu \leq \sqrt{2}$; so after removing the max, distributing, and cancelling, we have

$$\ddot{\varphi}(\nu) \leq \frac{-\frac{2}{3\pi} + \frac{83}{24\pi^3} \nu}{\left(1 - \frac{1}{4}\nu^2\right)^{3/2}}, \quad 0 < \nu \leq \sqrt{2}.$$

The denominator of this last expression is nonnegative and has singularities at $\pm 2$, and is clearly even symmetric; so it is maximized on $0 < \nu \leq \sqrt{2}$ at $\sqrt{2}$, and we have

$$\ddot{\varphi}(\nu) \leq \sqrt{8} \left(-\frac{2}{3\pi} + \frac{83}{24\pi^3} \nu\right), \quad 0 < \nu \leq \sqrt{2}.$$

Taking limits $\nu \searrow 0$, we can assert this bound on $[0, \sqrt{2}]$, and the bound is clearly an increasing function of $\nu$, from which it follows

$$\sup_{\nu \in [0, \sqrt{2}]} \ddot{\varphi}(\nu) \leq \sqrt{8} \left( -\frac{2}{3\pi} + \frac{83\sqrt{2}}{24\pi^3} \right) \leq -0.15,$$

where the last inequality follows from a numerical estimate of the constants. On the other hand, when $\sqrt{2} < \nu \leq \pi/2$, we have from (E.85) that

$$\ddot{\varphi}(\nu) \leq -\frac{2}{3\pi}\nu^3 + \frac{83}{24\pi^3}\nu^4, \quad \sqrt{2} \leq \nu \leq \pi/2.$$

If we differentiate the degree four polynomial on the RHS of this bound and solve for critical points, we find a double critical point at $\nu = 0$ and a critical point at $\nu = 12\pi^2/83$; a numerical estimate confirms that this critical point lies in the interior of $[\sqrt{2}, \pi/2]$. The second derivative of the RHS is $-(4/\pi)\nu + 83/(2\pi^3)\nu^2$, and plugging in $\nu = 12\pi^2/83$ gives a value of $-48\pi/83 + 144\pi/83$, which is positive; hence the RHS is maximized on the boundary, i.e.,

$$\ddot{\varphi}(\nu) \leq -\frac{2}{3\pi}\nu^3 + \frac{83}{24\pi^3}\nu^4 \leq \max\left\{ -\frac{2^{5/2}}{3\pi} + \frac{83}{6\pi^3}, -\frac{\pi^2}{12} + \frac{83\pi}{384} \right\}, \quad \sqrt{2} \leq \nu \leq \pi/2.$$

A numerical estimate shows that the RHS of the last inequality is no larger than $-0.14$. Since the intervals we have proved a bound over cover $[0, \pi/2]$, this proves the claim with $c = -0.14$.

The bound $\dot{\varphi} < 1$ on $(0, \pi)$ follows from the fact that $\varphi$ is strictly concave on $(0, \pi)$ and the mean value theorem; we have already shown $\dot{\varphi} > 0$ in proving strict increasingness of $\varphi$. Similarly, the proof of strict concavity in the interior has already established $\ddot{\varphi} < 0$. To obtain the lower bound on $\ddot{\varphi}$, we use that $\ddot{\varphi}$ is continuous on $[0, \pi]$ and the Weierstrass theorem to assert that there is $C \geq 0$ such that $\ddot{\varphi} \geq -C$ on $[0, \pi]$; because $\ddot{\varphi}(0) \neq 0$, we actually have $C > 0$.

For the quadratic model, we use our previous results and Taylor expand $\varphi$ about 0; we get immediately

$$\varphi(\nu) \geq \nu + \nu^2 \frac{\inf_{\nu \in [0, \pi]} \ddot{\varphi}(\nu)}{2} \geq \nu - (C/2)\nu^2.$$

For the upper bound, we can assert immediately on $[0, \pi/2]$ a bound

$$\varphi(\nu) \leq \nu - c\nu^2,$$

where $c = 0.07$ suffices. To extend the bound to $\nu \in [\pi/2, \pi]$, we employ a bootstrapping argument; because $\varphi$ is concave, we have a bound

$$\varphi(\nu) \leq \varphi(\pi/2) + \dot{\varphi}(\pi/2)(\nu - \pi/2)$$
$$= \cos^{-1} \pi^{-1} + \frac{\pi/2}{\sqrt{\pi^2 - 1}} (\nu - \pi/2),$$

where the second line plugs into the formulas for $\varphi$ and $\dot{\varphi}$. We will show that the graph of $\nu - c\nu^2$ lies entirely above the graph of the RHS of this inequality. This condition is equivalent to

$$-c\nu^2 + \left( 1 - \frac{\pi/2}{\sqrt{\pi^2 - 1}} \right) \nu + \left( \frac{(\pi/2)^2}{\sqrt{\pi^2 - 1}} - \cos^{-1} \pi^{-1} \right) \geq 0;$$

the LHS of this inequality is a concave quadratic with maximizer $\nu_\star = 1/(2c)\left(1 - \frac{\pi/2}{\sqrt{\pi^2-1}}\right)$, and numerical estimation of the constants gives $\nu_\star \geq \pi$. Since $\nu_\star$ is outside $[\pi/2, \pi]$ and the quadratic is concave, we conclude that the bound is tightest at the boundary point $\pi/2$, and one checks numerically

$$-c\pi^2/4 + \left( 1 - \frac{\pi/2}{\sqrt{\pi^2 - 1}} \right) \pi/2 + \left( \frac{(\pi/2)^2}{\sqrt{\pi^2 - 1}} - \cos^{-1} \pi^{-1} \right) \geq 0.15 > 0,$$

which establishes that the bound $\varphi(\nu) \leq \nu - c\nu^2$ actually holds on all of $[0, \pi]$. This completes the proof of all of the claims. $\qquad \square$

# F  CONTROLLING CHANGES DURING TRAINING

## F.1  PRELIMINARIES

We now consider the changes in the integral operator $\Theta_k$ during gradient descent. In this section we restore the iteration subscript (that is dropped in other sections to lighten notation) to various quantities. $\Theta_k$ changes during training as a result of both smooth changes in the features at all layers and non-smooth changes in the backward features $\{\boldsymbol{\beta}_t^\ell(\boldsymbol{x})\}$ due to the non-smoothness of the derivative of the ReLU function.

Because of the difficulty of reasoning precisely about the changes in $\Theta_t$, we will bound these rather naively by controlling $\Theta_t$ over all possible support patterns of the features given a bound on the norm change of the pre-activations.

We now define a trajectory in parameter space that interpolates between the iterates of gradient descent, given for any $k' \in \{0, \ldots, k\}$ and $s \in [0, 1]$ by

$$\boldsymbol{\theta}_{k'+s}^N = \boldsymbol{\theta}_{k'}^N - \tau s \tilde{\boldsymbol{\nabla}} \mathcal{L}^N(\boldsymbol{\theta}_{k'}^N), \tag{F.1}$$

(with the formal derivative $\tilde{\boldsymbol{\nabla}}$ defined in Appendix A.1). We will henceforth use $k'$ to denote an integer indexing the iteration number and $t$ to denote a continuous parameter taking values in $[0, k]$ (such that $k' = \lfloor t \rfloor$, $s = t - \lfloor t \rfloor$). Quantities indexed by $t$ are ones where the parameters take the value $\boldsymbol{\theta}_t^N$. To lighten notation, we will drop the $N$ superscript when referring to time-indexed quantities (aside from $\zeta_k^N$ and $\Theta_k^N$), but all such quantities depend on the parameters as defined by (F.1).

Instead of considering the change in the features $\{\boldsymbol{\alpha}_t^\ell(\boldsymbol{x})\}$ directly, it will be more convenient to work in terms of the pre-activations, which are given at layer $\ell$ by

$$\boldsymbol{\rho}_t^\ell(\boldsymbol{x}) = \boldsymbol{W}_t^\ell \boldsymbol{P}_{I_{\ell-1,t}(\boldsymbol{x})} \boldsymbol{W}_t^{\ell-1} \boldsymbol{P}_{I_{\ell-2,t}(\boldsymbol{x})} \boldsymbol{W}_t^{\ell-2} \ldots \boldsymbol{P}_{I_{1,t}(\boldsymbol{x})} \boldsymbol{W}_t^1 \boldsymbol{x}.$$

We define a maximal allowable change in the pre-activation norm by

$$\eta = \frac{C_\eta L^{3/2+q}}{\sqrt{n}} \tag{F.2}$$

for $q \geq 0$ and a constant $C_\eta$ to be specified later, where the scaling is chosen with foresight. We can then define a maximal number of iterations such that the pre-activation norms at all layers along the trajectory (F.1) change by no more than $\eta$. This number $k_\eta$ must satisfy

$$\sup_{t \in [0, k_\eta], \boldsymbol{x} \in \mathcal{M}, \ell \in [L]} \left\| \boldsymbol{\rho}_t^\ell(\boldsymbol{x}) - \boldsymbol{\rho}_0^\ell(\boldsymbol{x}) \right\|_2 \leq \eta \tag{F.3}$$

for $\eta$ given by (F.2). Our goal will be to show that we can in fact train for long enough so as to reduce the fitting error without exceeding $k_\eta$ iterations.

## F.2  CHANGES IN FEATURE SUPPORTS DURING TRAINING

Recalling the definition of the feature supports at layer $\ell$, time $t$ and $\boldsymbol{x} \in \mathcal{M}$ by $I_{\ell,t}(\boldsymbol{x}) = \text{supp}(\boldsymbol{\alpha}_t^\ell(\boldsymbol{x}) > \boldsymbol{0}) \subseteq [n]$, we denote by $\mathcal{I}_t(\boldsymbol{x}) = (I_{1,t}(\boldsymbol{x}), \ldots, I_{L,t}(\boldsymbol{x}))$ the collection of these support patterns at all layers. We would next like to relate the smooth changes in the pre-activation norms to the non-smooth changes in the supports of the features. We denote by $\mathcal{J} = (J_1, \ldots, J_L)$ a collection of support patterns with $J_i \in [n]$. We now consider sets of support patterns that are not too different from those at initialization, as defined by

$$\mathcal{B}(\boldsymbol{y}, \eta) = \{\text{supp}(\boldsymbol{y} + \boldsymbol{v} > \boldsymbol{0}) \mid \|\boldsymbol{v}\|_2 \leq \eta\}, \tag{F.4}$$

$$\overline{\mathcal{J}}_\eta(\boldsymbol{x}) = \underset{\ell \in [L]}{\otimes} \mathcal{B}(\boldsymbol{\rho}_0^\ell(\boldsymbol{x}), \eta), \tag{F.5}$$

$$\overline{\mathcal{J}}_\eta(\mathcal{M}) = \bigcup_{\boldsymbol{x} \in \mathcal{M}} \overline{\mathcal{J}}_\eta(\boldsymbol{x}). \tag{F.6}$$

Note that $\mathcal{B}(\boldsymbol{\rho}_0^\ell(\boldsymbol{x}), \eta)$ is simply the set of supports of the positive entries of $\boldsymbol{\rho}_0^\ell(\boldsymbol{x}) + \boldsymbol{v}$ for every possible perturbation $\boldsymbol{v}$ of norm at most $\eta$. We consider all possible perturbations due to the complex

nature of the training dynamics. As a result of this worst-casing, the scaling we will require of the depth and width of the network in order to guarantee that the changes during training are sufficiently small is expected to be suboptimal.

For a given general support pattern $\mathcal{J}$, we define generalized backward features and transfer matrices

$$
\begin{aligned}
\boldsymbol{\beta}_{\mathcal{J}t}^{\ell} &= \left(\boldsymbol{W}_t^{NL+1}\boldsymbol{P}_{J_L}\boldsymbol{W}_t^{NL}\dots\boldsymbol{W}_t^{N\ell+2}\boldsymbol{P}_{J_{\ell+1}}\right)^*, \\
\tilde{\boldsymbol{\Gamma}}_{\mathcal{J}t}^{\ell:\ell'} &= \boldsymbol{W}_t^{\ell}\boldsymbol{P}_{J_{\ell-1}}\boldsymbol{W}_t^{N\ell-1}\dots\boldsymbol{P}_{J_{\ell'}}\boldsymbol{W}_t^{N\ell'}
\end{aligned}
\tag{F.7}
$$

where the weights are given by (F.1) (and thus $\boldsymbol{\beta}_t^{\ell}(\boldsymbol{x}) = \boldsymbol{\beta}_{\mathcal{I}_t(\boldsymbol{x})t}^{N\ell}$). By controlling these objects for every possible set of supports $\mathcal{J}$ that can be encountered during training, we can control the smooth changes in the features themselves. A first step towards this end is understanding how many such support patterns we expect to see given the constraint in (F.2).

In order to bound the number supports that can be encountered during training, we need to control the diameter of $\mathcal{B}(\boldsymbol{\rho}_0^{\ell}(\boldsymbol{x}),\eta)$. This can be done by defining

$$
\delta_{\eta}(\boldsymbol{\rho}_0^{\ell}(\boldsymbol{x})) = \max_{\|\boldsymbol{v}\|_2\leq\eta}\left|\text{supp}\left(\boldsymbol{\rho}_0^{\ell}(\boldsymbol{x}) > \boldsymbol{0}\right) \ominus \text{supp}\left(\boldsymbol{\rho}_0^{\ell}(\boldsymbol{x}) + \boldsymbol{v} > \boldsymbol{0}\right)\right| \tag{F.8}
$$

where $\ominus$ denotes the symmetric difference. Since the pre-activation at a given layer are Gaussian variables conditioned on all the previous layer weights, bounding the size of $\delta_{\eta}(\boldsymbol{\rho}_0^{\ell}(\boldsymbol{x}))$ can be reduced to showing concentration of a certain function of Gaussian order statistics. This is achieved in the following lemma :

**Lemma F.1.** *For $\eta$ given by* (F.2)*, if $n, L, d$ satisfy the requirements of lemma F.6 and $n > d^5$ for some constant $K$, then for a vector $\boldsymbol{g} \in \mathbb{R}^n$, $g_i \sim_{iid} \mathcal{N}(0, \frac{1}{n})$ we have*

$$
\mathbb{P}\left[\delta_{\eta}(\boldsymbol{g}) > Cn\eta^{2/3}\right] \leq C'e^{-cd}
$$

*for some constants $c, C, C'$.*

*Proof.* Let

$$
|\boldsymbol{g}|_{(1)} \leq |\boldsymbol{g}|_{(2)} \leq \cdots \leq |\boldsymbol{g}|_{(n)}
$$

denote the order statistics of the magnitudes of the elements of $\boldsymbol{g}$. We will show that bounding $\delta_{\eta}(\boldsymbol{g})$ can be reduced to understanding what is the smallest $k$ such that

$$
|\boldsymbol{g}|_{(1)}^2 + \cdots + |\boldsymbol{g}|_{(k)}^2 \geq \eta^2.
$$

We denote this value of $k$ by $k_{\eta}$. Define indices $j_i$ by $|g_{J_i}| = |g|_{(i)}$ (and breaking ties arbitrarily in case several order statistics are equal). To see that

$$
k_{\eta} - 1 \leq \delta_{\eta}(\boldsymbol{g}) \leq k_{\eta} \tag{F.9}
$$

it suffices to note that since $\sum\limits_{i=1}^{k_{\eta}-1}|\boldsymbol{g}|_{(k)}^2 < \eta^2$ one can choose $\varepsilon > 0$ small enough such that

$$
\boldsymbol{y} = \boldsymbol{g} - \sum_{i=1}^{k_{\eta}-1}(1+\varepsilon)g_{J_i}\boldsymbol{e}_{J_i} \in \mathbb{B}_E(\boldsymbol{g},\eta),
$$

which will give $d_s(\boldsymbol{g},\boldsymbol{y}) = k_{\eta} - 1 \Rightarrow \delta_{\eta} \geq k_{\eta} - 1$. To prove the second inequality, consider $\boldsymbol{y} = \boldsymbol{g} - \sum\limits_{i=1}^{\delta_{\eta}}g_{J_i}\boldsymbol{e}_{J_i}$. Clearly for any $\boldsymbol{y}'$ such that $d_s(\boldsymbol{g},\boldsymbol{y}') = \delta_{\eta}$ we have $\|\boldsymbol{g}-\boldsymbol{y}\|_2 \leq \|\boldsymbol{g}-\boldsymbol{y}'\|_2$. Since there exists at least one such $\boldsymbol{y}' \in \mathbb{B}_E(\boldsymbol{g},\eta)$, it follows that $\|\boldsymbol{g}-\boldsymbol{y}\|_2 \leq \|\boldsymbol{g}-\boldsymbol{y}'\|_2 \leq \eta$, and hence the smallest $k$ such that $\sum\limits_{i=1}^{k}\left|g_{(i)}\right|^2 \geq \eta^2$ must obey $k \geq \delta_{\eta}$.

For $\eta$ defined in (F.2), we can satisfy the requirement on $\eta$ in lemma F.6 by requiring $n > d^5$ for some $K$. Applying this lemma, we find that there is a constant $K'$ such that for $k = \lceil K'n\eta^{2/3}\rceil$ we have

$$
\mathbb{P}\left[\sum_{i=1}^{k}|g|_{(i)}^2 \geq \eta^2\right] \geq 1 - C'e^{-cd}
$$

from which it follows immediately that

$$\mathbb{P}\left[k_\eta \leq \left\lceil K' n \eta^{2/3} \right\rceil \right] \geq 1 - C' e^{-cd}.$$

Choosing some constant $C$ such that $\left\lceil K' n \eta^{2/3} \right\rceil \leq C n \eta^{2/3}$ and using (F.9) allows us to bound $\delta_\eta(\boldsymbol{g})$ with the same probability. $\qquad\square$

With this result in hand, we can control the objects in (F.7) for all the supports in $\overline{\mathcal{J}}_\eta(\mathcal{M})$.

**Lemma F.2.** *Assume $d, L, n$ satisfy the assumptions of lemmas D.2, F.1, D.8, D.14 and additionally*

$$n \geq \max \left\{ K d L^{9+2q}, K' \left(\log n\right)^{3/2}, C_0^3 C_\eta^2 L^{6+2q} \right\},$$

*for some constants $K, K', C_0$, where $q$ is the constant in (F.2).*

*Then*

*i) for $\eta$, $\overline{\mathcal{J}}_\eta(\boldsymbol{x})$ defined in (F.2),(F.5), on an event of probability at least $1 - e^{-cd}$, simultaneously*

$$\sup_{\substack{\boldsymbol{x} \in \mathcal{M}}} \sup_{\substack{\mathcal{J} \in \overline{\mathcal{J}}_\eta(\boldsymbol{x})}} \sup_{\substack{\ell \in [L] \\ 1 \leq \ell' < \ell}} \left\| \boldsymbol{\rho}_0^\ell(\boldsymbol{x}) \right\|_2 \leq C_2,$$

$$\sup_{\substack{\boldsymbol{x} \in \mathcal{M}}} \sup_{\substack{\mathcal{J} \in \overline{\mathcal{J}}_\eta(\boldsymbol{x})}} \sup_{\substack{\ell \in [L] \\ 1 \leq \ell' < \ell}} \left\| \boldsymbol{\beta}_{\mathcal{J},0}^{\ell-1} \right\|_2 \leq C_2 \sqrt{n},$$

$$\sup_{\substack{\boldsymbol{x} \in \mathcal{M}}} \sup_{\substack{\mathcal{J} \in \overline{\mathcal{J}}_\eta(\boldsymbol{x})}} \sup_{\substack{\ell \in [L] \\ 1 \leq \ell' < \ell}} \left\| \tilde{\boldsymbol{\Gamma}}_{\mathcal{J},0}^{\ell:\ell'} \right\| \leq C_2 \sqrt{L}.$$

*ii) for $T_\eta$ defined in (F.3), on an event of probability at least $1 - e^{-cd}$,*

$$\sup_{\substack{\boldsymbol{x} \in \mathcal{M}, t \in [0, T_\eta], \ell \in [L]}} \left\| \boldsymbol{\beta}_{\mathcal{I}_0(\boldsymbol{x}),0}^{\ell-1} - \boldsymbol{\beta}_{\mathcal{I}_t(\boldsymbol{x}),0}^{\ell-1} \right\|_2 \leq C C_\eta^{2/3} \log^{3/4}(L) d^{3/4} L^{3+2q/3} n^{5/12}.$$

*for some constants $c, C$.*

*Proof.* Deferred to F.5. $\qquad\square$

## F.3 Changes in Features During Training

We can now bound the smooth changes during training:

**Lemma F.3** (Smooth changes during training). *Set the step size $\tau$ and a bound on the maximal number of iterations $k_{\max}$ such that*

$$k_{\max} \tau = \frac{L^q}{n}$$

*for some constant $q$. Assume $n, L, d$ satisfy the requirements of lemmas F.2, and in particular $n \geq K d L^{9+2q}$ for some $K$. Assume also that given some $k \leq k_{\max} - 1$, for all $k' \in \{0, \ldots, k\}$,*

$$\left\| \zeta_{k'}^N \right\|_{L^2_{\mu^N}} \leq C \sqrt{d}. \tag{F.10}$$

*Then on an event of probability at least $1 - e^{-cd}$, one has simultaneously*

$$\sup_{\substack{\boldsymbol{x} \in \mathcal{M}, \ell \in [L], k' \in \{0, \ldots, k+1\}}} \left\| \boldsymbol{\rho}_{k'}^\ell(\boldsymbol{x}) - \boldsymbol{\rho}_0^\ell(\boldsymbol{x}) \right\|_2 \leq C' L^{3/2+q} \sqrt{\frac{d}{n}},$$

$$\sup_{\substack{\boldsymbol{x} \in \mathcal{M}, \ell \in [L], k' \in \{0, \ldots, k+1\}}} \left( \left\| \boldsymbol{\beta}_{k'}^{\ell-1}(\boldsymbol{x}) - \boldsymbol{\beta}_0^{\ell-1}(\boldsymbol{x}) \right\|_2 - \left\| \boldsymbol{\beta}_{\mathcal{I}_{k'}0}^{\ell-1}(\boldsymbol{x}) - \boldsymbol{\beta}_{\mathcal{I}_0 0}^{\ell-1}(\boldsymbol{x}) \right\|_2 \right) \leq C' \sqrt{d} L^{3/2+q},$$

*for some constants $c, C, C'$.*

*Proof.* We will bound the smooth changes in the network features during gradient descent with respect to either the population measure $\mu^\infty$ or the finite sample measure $\mu^N$. We denote a measure that can be one of these two by $\mu^N$.

For any collection of supports $\mathcal{J} \in \overline{\mathcal{J}}_\eta(\mathcal{M})$, define generalized backward features and transfer matrices at $t$ by $\boldsymbol{\beta}_{\mathcal{J}t}^\ell, \tilde{\boldsymbol{\Gamma}}_{\mathcal{J}t}^{\ell:\ell'}$. These are obtained by setting the network parameters to be $\boldsymbol{\theta}_t^N$ according to (F.1), but setting all the support patterns to be those in $\mathcal{J}$.

We then define for any $t \in [0, k+1]$,

$$
\begin{aligned}
\overline{\rho}_t^\eta &= \sup_{\boldsymbol{x} \in \mathcal{M}, \ell \in [L], t' \in [0,t]} \left\| \boldsymbol{\rho}_{t'}^\ell(\boldsymbol{x}) - \boldsymbol{\rho}_0^\ell(\boldsymbol{x}) \right\|_2 + \sup_{\boldsymbol{x} \in \mathcal{M}, \ell \in [L]} \left\| \boldsymbol{\rho}_0^\ell(\boldsymbol{x}) \right\|_2, \\
\overline{\beta}_t^\eta &= \sup_{\ell \in [L], \mathcal{J} \in \overline{\mathcal{J}}_\eta(\mathcal{M}), t' \in [0,t]} \left\| \boldsymbol{\beta}_{\mathcal{J}t'}^\ell - \boldsymbol{\beta}_{\mathcal{J}0}^\ell \right\|_2 + \sup_{\ell \in [L], \mathcal{J} \in \overline{\mathcal{J}}_\eta(\mathcal{M})} \left\| \boldsymbol{\beta}_{\mathcal{J}0}^\ell \right\|_2, \\
\overline{\Gamma}_t^\eta &= \sup_{\ell' \leq \ell \in [L], \mathcal{J} \in \overline{\mathcal{J}}_\eta(\mathcal{M}), t' \in [0,t]} \left\| \tilde{\boldsymbol{\Gamma}}_{\mathcal{J}t'}^{\ell:\ell'} - \tilde{\boldsymbol{\Gamma}}_{\mathcal{J}0}^{\ell:\ell'} \right\| + \sup_{\ell' \leq \ell \in [L], \mathcal{J} \in \overline{\mathcal{J}}_\eta(\mathcal{M})} \left\| \tilde{\boldsymbol{\Gamma}}_{\mathcal{J}0}^{\ell:\ell'} \right\|.
\end{aligned}
\tag{F.11}
$$

We have for all $k' \leq k+1$,

$$
\left\| \boldsymbol{\rho}_{k'}^\ell(\boldsymbol{x}) - \boldsymbol{\rho}_0^\ell(\boldsymbol{x}) \right\|_2 \leq \overline{\rho}_{k'}^\eta - \overline{\rho}_0^\eta,
\tag{F.12}
$$

while

$$
\begin{aligned}
\left\| \boldsymbol{\beta}_{k'}^\ell(\boldsymbol{x}) - \boldsymbol{\beta}_0^\ell(\boldsymbol{x}) \right\|_2 &= \left\| \boldsymbol{\beta}_{\mathcal{I}_{k'}(\boldsymbol{x}),k'}^\ell - \boldsymbol{\beta}_{\mathcal{I}_0(\boldsymbol{x}),0}^\ell \right\|_2 \\
&\leq \left\| \boldsymbol{\beta}_{\mathcal{I}_{k'}(\boldsymbol{x}),k'}^\ell - \boldsymbol{\beta}_{\mathcal{I}_{k'}(\boldsymbol{x}),0}^\ell \right\|_2 + \left\| \boldsymbol{\beta}_{\mathcal{I}_{k'}(\boldsymbol{x}),0}^\ell - \boldsymbol{\beta}_{\mathcal{I}_0(\boldsymbol{x}),0}^\ell \right\|_2 \\
&\leq \overline{\beta}_{k'}^\eta - \overline{\beta}_0^\eta + \left\| \boldsymbol{\beta}_{\mathcal{I}_{k'}(\boldsymbol{x}),0}^\ell - \boldsymbol{\beta}_{\mathcal{I}_0(\boldsymbol{x}),0}^\ell \right\|_2.
\end{aligned}
\tag{F.13}
$$

It follows that we can control the difference norms of the pre-activations and backward features by controlling the magnitudes of $\overline{\rho}_{k'}^\eta, \overline{\beta}_{k'}^\eta$. In order to control these, we also note that for any $t \in [0, k+1]$,

$$
\left\| \boldsymbol{\alpha}_t^\ell(\boldsymbol{x}) \right\|_2 \leq \left\| \boldsymbol{\rho}_t^\ell(\boldsymbol{x}) \right\|_2 \leq \left\| \boldsymbol{\rho}_t^\ell(\boldsymbol{x}) - \boldsymbol{\rho}_0^\ell(\boldsymbol{x}) \right\|_2 + \left\| \boldsymbol{\rho}_0^\ell(\boldsymbol{x}) \right\|_2 \leq \overline{\rho}_t^\eta,
\tag{F.14}
$$

and similarly

$$
\begin{aligned}
\left\| \boldsymbol{\beta}_{\mathcal{J}t}^\ell \right\|_2 &\leq \overline{\beta}_t^\eta, \\
\left\| \tilde{\boldsymbol{\Gamma}}_{\mathcal{J}t}^{\ell:\ell'} \right\| &\leq \overline{\Gamma}_t^\eta.
\end{aligned}
\tag{F.15}
$$

In particular, the above bounds hold when $\mathcal{J} = \mathcal{I}_t(\boldsymbol{x})$. We would now like to understand how the quantities $\left( \overline{\rho}_{k'}^\eta, \overline{\beta}_{k'}^\eta, \overline{\Gamma}_{k'}^\eta \right)$ evolve under gradient descent. Towards this end, for any $k' \in \{0, \ldots, k\}$ and $s \in [0,1]$ we compute at any point of differentiability

$$
\begin{aligned}
&\frac{\tilde{\partial}}{\partial s} \boldsymbol{\rho}_{k'+s}^\ell(\boldsymbol{x}) \\
={}& -\tau \left. \frac{\tilde{\partial} \boldsymbol{\rho}^\ell(\boldsymbol{x})}{\partial \boldsymbol{\theta}} \right|_{\boldsymbol{\theta}_{k'} - s\tau \tilde{\boldsymbol{\nabla}} \mathcal{L}^N(\boldsymbol{\theta}_{k'}^N)}^* \tilde{\boldsymbol{\nabla}} \mathcal{L}^N(\boldsymbol{\theta}_{k'}^N) \\
={}& -\tau \left( \frac{\tilde{\partial} \boldsymbol{\rho}_{k'+s}^\ell(\boldsymbol{x})}{\partial \boldsymbol{\theta}} \right)^* \tilde{\boldsymbol{\nabla}} \mathcal{L}^N(\boldsymbol{\theta}_{k'}^N) \\
={}& -\tau \sum_{i=1}^\ell \sum_{j=1}^{n_i} \sum_{l=1}^{n_{i-1}} \frac{\tilde{\partial} \boldsymbol{\rho}_{k'+s}^\ell(\boldsymbol{x})}{\partial W_{jl}^i} \frac{\tilde{\partial} \mathcal{L}^N(\boldsymbol{\theta}_{k'}^N)}{\partial W_{jl}^i} \\
={}& -\tau \sum_{i=1}^\ell \int_{\boldsymbol{x}' \in \mathcal{M}} \langle \boldsymbol{\alpha}_{k'+s}^{i-1}(\boldsymbol{x}), \boldsymbol{\alpha}_{k'}^{i-1}(\boldsymbol{x}') \rangle \tilde{\boldsymbol{\Gamma}}_{k'+s}^{\ell:i+1}(\boldsymbol{x}) \boldsymbol{P}_{I_{i,k'+s}(\boldsymbol{x})} \boldsymbol{\beta}_{k'}^{i-1}(\boldsymbol{x}') \zeta_{k'}^N(\boldsymbol{x}') \mathrm{d}\mu^N(\boldsymbol{x}').
\end{aligned}
$$

Using (F.14) and (F.15) then gives

$$
\begin{aligned}
\left\|\frac{\tilde{\partial}}{\partial s}\boldsymbol{\rho}_{k'+s}^{\ell}(\boldsymbol{x})\right\|_2 \quad &\leq \tau L \left(\overline{\rho}_{k'+s}^{\eta}\right)^2 \overline{\beta}_{k'+s}^{\eta} \overline{\Gamma}_{k'+s}^{\eta} \int_{\boldsymbol{x}'\in\mathcal{M}} \left|\zeta_{k'}^{N}(\boldsymbol{x}')\right| \mathrm{d}\mu^{N}(\boldsymbol{x}') \\
&\leq \tau L \left(\overline{\rho}_{k'+s}^{\eta}\right)^2 \overline{\beta}_{k'+s}^{\eta} \overline{\Gamma}_{k'+s}^{\eta} \left\|\zeta_{k'}^{N}\right\|_{L^2_{\mu^N}} \\
&\leq C\sqrt{d}\tau L \left(\overline{\rho}_{k'+s}^{\eta}\right)^2 \overline{\beta}_{k'+s}^{\eta} \overline{\Gamma}_{k'+s}^{\eta},
\end{aligned}
$$

where we used Jensen's inequality in the second line and our assumption that the error up to iteration $k$ has bounded $L^2_{\mu^N}$ norm, and we additionally assumed $\overline{\rho}_t^{\eta}, \overline{\beta}_t^{\eta}, \overline{\Gamma}_t^{\eta} \geq 1$. Arguing as in the proof of Lemma B.8 for absolute continuity, it follows that

$$
\begin{aligned}
\left\|\boldsymbol{\rho}_t^{\ell}(\boldsymbol{x}) - \boldsymbol{\rho}_0^{\ell}(\boldsymbol{x})\right\|_2 \quad &\leq \sum_{k'=0}^{\lfloor t\rfloor-1} \left\|\boldsymbol{\rho}_{k'+1}^{\ell}(\boldsymbol{x}) - \boldsymbol{\rho}_{k'}^{\ell}(\boldsymbol{x})\right\|_2 + \left\|\boldsymbol{\rho}_t^{\ell}(\boldsymbol{x}) - \boldsymbol{\rho}_{\lfloor t\rfloor}^{\ell}(\boldsymbol{x})\right\|_2 \\
&= \sum_{k'=0}^{\lfloor t\rfloor-1} \left\|\int_0^1 \frac{\tilde{\partial}}{\partial s}\boldsymbol{\rho}_{k'+s}^{\ell}(\boldsymbol{x})\mathrm{d}s\right\|_2 + \left\|\int_0^{t-\lfloor t\rfloor} \frac{\tilde{\partial}}{\partial s}\boldsymbol{\rho}_{\lfloor t\rfloor+s}^{\ell}(\boldsymbol{x})\mathrm{d}s\right\|_2 \\
&\leq \sum_{k'=0}^{\lfloor t\rfloor-1} \int_0^1 \left\|\frac{\tilde{\partial}}{\partial s}\boldsymbol{\rho}_{k'+s}^{\ell}(\boldsymbol{x})\right\|_2 \mathrm{d}s + \int_0^{t-\lfloor t\rfloor} \left\|\frac{\tilde{\partial}}{\partial s}\boldsymbol{\rho}_{\lfloor t\rfloor+s}^{\ell}(\boldsymbol{x})\right\|_2 \mathrm{d}s \\
&\leq C\sqrt{d}\tau t L \left(\overline{\rho}_t^{\eta}\right)^2 \overline{\beta}_t^{\eta} \overline{\Gamma}_t^{\eta}
\end{aligned}
\tag{F.16}
$$

Since the above holds for all choices of $\boldsymbol{x}, \ell$ simultaneously, we conclude that

$$
\overline{\rho}_t^{\eta} - \overline{\rho}_0^{\eta} \leq C\sqrt{d}\tau t L \left(\overline{\rho}_t^{\eta}\right)^2 \overline{\beta}_t^{\eta} \overline{\Gamma}_t^{\eta}.
\tag{F.17}
$$

An analogous calculation for the other quantities in (F.11) gives the following set of coupled difference inequalities:

$$
\begin{pmatrix} \overline{\rho}_t^{\eta} - \overline{\rho}_0^{\eta} \\ \overline{\beta}_t^{\eta} - \overline{\beta}_0^{\eta} \\ \overline{\Gamma}_t^{\eta} - \overline{\Gamma}_0^{\eta} \end{pmatrix} \leq C\sqrt{d}L\tau t\overline{\rho}_t^{\eta}\overline{\beta}_t^{\eta}\overline{\Gamma}_t^{\eta} \begin{pmatrix} \overline{\rho}_t^{\eta} \\ \overline{\beta}_t^{\eta} \\ \overline{\Gamma}_t^{\eta} \end{pmatrix}.
\tag{F.18}
$$

Instead of solving (F.18), we obtain sufficient control by defining $k^*$ s.t.

$$
\forall t \in [0, k^*]: \quad \overline{\rho}_t^{\eta} \leq 2\overline{\rho}_0^{\eta} \quad \wedge \quad \overline{\beta}_t^{\eta} \leq 2\overline{\beta}_0^{\eta} \quad \wedge \quad \overline{\Gamma}_t^{\eta} \leq 2\overline{\Gamma}_0^{\eta}.
\tag{F.19}
$$

For any $t \in [0, k^*]$, we obtain a sufficient condition for satisfying the above constraint using (F.18), namely

$$
\begin{aligned}
\overline{\rho}_0^{\eta} + C'\sqrt{d}\tau t L \left(\overline{\rho}_0^{\eta}\right)^2 \overline{\beta}_0^{\eta}\overline{\Gamma}_0^{\eta} \quad &\leq 2\overline{\rho}_0^{\eta} \\
\Leftrightarrow \tau t \quad &\leq \frac{1}{C'\sqrt{d}L\overline{\rho}_0^{\eta}\overline{\beta}_0^{\eta}\overline{\Gamma}_0^{\eta}}
\end{aligned}
\tag{F.20}
$$

for some constant $C'$. Using the bounds for $\overline{\beta}_t^{\eta} - \overline{\beta}_0^{\eta}$ and $\overline{\Gamma}_t^{\eta} - \overline{\Gamma}_0^{\eta}$ in (F.18) to satisfy the second and third condition in (F.19) gives an identical constraint on $\tau t$.

In order to control these quantities at $t = 0$ we define an event

$$
\mathcal{G} = \bigcap_{\substack{\boldsymbol{x}\in\mathcal{M}, \\ \mathcal{J}\in\overline{\mathcal{J}}_{\eta}(\boldsymbol{x}), \\ 1\leq\ell'\leq\ell\in[L]}} \begin{aligned} &\left\{\left\|\boldsymbol{\rho}_0^{\ell}(\boldsymbol{x})\right\|_2 \leq C_2\right\} \\ \cap&\left\{\left\|\boldsymbol{\beta}_{\mathcal{J},0}^{\ell-1}\right\|_2 \leq C_2\sqrt{n}\right\} \\ \cap&\left\{\left\|\tilde{\boldsymbol{\Gamma}}_{\mathcal{J},0}^{\ell:\ell'}\right\| \leq C_2\sqrt{L}\right\}, \end{aligned}
\tag{F.21}
$$

the probability of which can be controlled using lemma F.2. On $\mathcal{G}$, the upper bound on $\tau t$ in (F.20) is at least

$$
\frac{1}{C'\sqrt{d}L\overline{\rho}_0^{\eta}\overline{\beta}_0^{\eta}\overline{\Gamma}_0^{\eta}} \geq \frac{1}{C''\sqrt{d}L^{3/2}\sqrt{n}}
\tag{F.22}
$$

for some $C''$.

We would now like to pick $\tau k_{\max}$, and ensure that any $t \in [0, k+1]$ satisfies the constraint above if $k+1 \leq k_{\max}$. The analysis also assumes that $\tau k_{\max} \leq T_\eta$ for which (F.3) holds. We will then pick the scaling factor $C_\eta$ for the pre-activation norm bound in (F.2) in order to satisfy that constraint as well. We choose

$$\tau k_{\max} = \frac{L^q}{n}. \tag{F.23}$$

In order to ensure that $k_{\max} \leq k^*$ holds, we use (F.20) and (F.22), and require

$$\frac{L^q}{n} \leq \frac{1}{C''\sqrt{d}L^{3/2}\sqrt{n}}$$

which is satisfied by demanding $n \geq KdL^{3+2q}$ for some constant $K$. Using (F.17) and (F.22), on $\mathcal{G}$ we have

$$\sup_{t\in[0,k+1],\boldsymbol{x}\in\mathcal{M},\ell\in[L]} \left\|\boldsymbol{\rho}_t^\ell(\boldsymbol{x}) - \boldsymbol{\rho}_0^\ell(\boldsymbol{x})\right\|_2 \quad \leq \bar{\rho}_t^\eta - \bar{\rho}_0^\eta$$

$$\leq C'\tau t\sqrt{d}L\left(\bar{\rho}_0^\eta\right)^2\bar{\beta}_0^\eta\bar{\Gamma}_0^\eta$$

$$\leq C''\tau t\sqrt{d}L^{3/2}\sqrt{n}$$

$$\leq C''\tau k_{\max}\sqrt{d}L^{3/2}\sqrt{n}.$$

In order to ensure that $k_{\max} \leq k_\eta$ we therefore require

$$C''\tau k_{\max}\sqrt{d}L^{3/2}\sqrt{n} \leq \eta,$$

which using (F.2) and (F.23) is equivalent to

$$C''\frac{\sqrt{d}L^{q+3/2}}{\sqrt{n}} \leq C_\eta\frac{L^{q+3/2}}{\sqrt{n}},$$

and thus the constraint can be satisfied by choosing $C_\eta = C''\sqrt{d}$. Note that the constant $C_2$ in (F.21) (which enters $C''$) is set in lemma F.2 which takes $C_\eta$ as input (despite this, $C_2$ is independent of $C_\eta$). This lemma holds as long as $n \geq C_0^3 C_\eta^2 L^{6+2q}$, which we can guarantee by demanding $n \geq C_0^3\left(\sqrt{d}C''\right)^2 L^{6+2q}$.

We have thus ensured that our choice of $k_{\max}$ satisfies $k_{\max} \leq \min\{k^*, k_\eta\}$. We then obtain from the constraints in (F.19), the inequalities in (F.18) and the definition of $\mathcal{G}$, that on this event

$$\sup_{k'\in\{0,\ldots,k+1\}} \bar{\rho}_{k'}^\eta - \bar{\rho}_0^\eta \leq C''\tau k_{\max}\sqrt{d}L^{3/2}\sqrt{n} \leq C''\frac{\sqrt{d}L^{3/2+q}}{\sqrt{n}},$$

$$\sup_{k'\in\{0,\ldots,k+1\}} \bar{\beta}_{k'}^\eta - \bar{\beta}_0^\eta \leq C''\tau k_{\max}\sqrt{d}L^{3/2}n \leq C''\sqrt{d}L^{3/2+q}.$$

Then using (F.12) and (F.13), we obtain on an event of probability at least $1 - e^{-cd}$ simultaneously

$$\sup_{\boldsymbol{x}\in\mathcal{M},\,\ell\in[L],\,k'\in\{0,\ldots,k+1\}} \left\|\boldsymbol{\rho}_{k'}^\ell(\boldsymbol{x}) - \boldsymbol{\rho}_0^\ell(\boldsymbol{x})\right\|_2 \leq C'L^{3/2+q}\sqrt{\frac{d}{n}},$$

$$\sup_{\boldsymbol{x}\in\mathcal{M},\,\ell\in[L],\,k'\in\{0,\ldots,k+1\}} \left(\left\|\boldsymbol{\beta}_{k'}^{\ell-1}(\boldsymbol{x}) - \boldsymbol{\beta}_0^{\ell-1}(\boldsymbol{x})\right\|_2 - \left\|\boldsymbol{\beta}_{\mathcal{I}_{k'}0}^{\ell-1}(\boldsymbol{x}) - \boldsymbol{\beta}_{\mathcal{I}_00}^{\ell-1}(\boldsymbol{x})\right\|_2\right) \leq C'\sqrt{d}L^{3/2+q}.$$

$\square$

The combination of the last two lemmas allows us to control the changes in all the forward and backward features uniformly:

**Lemma F.4.** *Assume $n, L, d, k$ satisfy the requirements of lemmas F.2 and F.3, and additionally $n \geq KL^{36+8q}d^9$ for some $K$. Then one has simultaneously on an event of probability at least $1 - e^{-cd}$*

$$\sup_{\boldsymbol{x} \in \mathcal{M}, \, t \in [0,k+1], \, \ell \in [L]} \left\| \boldsymbol{\alpha}_t^\ell(\boldsymbol{x}) - \boldsymbol{\alpha}_0^\ell(\boldsymbol{x}) \right\|_2 \leq CL^{3/2+q}\sqrt{\frac{d}{n}},$$

$$\sup_{\boldsymbol{x} \in \mathcal{M}, \, t \in [0,k+1], \, \ell \in [L]} \left\| \boldsymbol{\beta}_t^{\ell-1}(\boldsymbol{x}) - \boldsymbol{\beta}_0^{\ell-1}(\boldsymbol{x}) \right\|_2 \leq C \log^{3/4}(L) d^{3/4} L^{3+2q/3} n^{5/12},$$

$$\sup_{\boldsymbol{x} \in \mathcal{M}, \, t \in [0,k+1], \, \ell \in [L]} \left\| \boldsymbol{\alpha}_t^\ell(\boldsymbol{x}) \right\|_2 \leq C,$$

$$\sup_{\boldsymbol{x} \in \mathcal{M}, \, t \in [0,k+1], \, \ell \in [L]} \left\| \boldsymbol{\beta}_t^{\ell-1}(\boldsymbol{x}) \right\|_2 \leq C\sqrt{n},$$

*for some constants $c, C$.*

*Proof.* Combine the results of lemmas F.2 and F.3 and take a union bound, using the triangle inequality to obtain the second two bounds. The assumption $n \geq KL^{36+8q}d^9$ is required in showing $\left\| \boldsymbol{\beta}_t^{\ell-1}(\boldsymbol{x}) \right\|_2 \leq C\sqrt{n}$. $\qquad\square$

### F.4 Changes in $\Theta_k^N$ During Training

With these results in hand, control of the changes in $\Theta_k^N$ during training is straightforward.

**Lemma F.5** (Uniform control of changes in $\Theta$ during training). *Denoting the gradient descent step size by $\tau$, choose some $k_{\max}$ such that*

$$k_{\max}\tau = \frac{L^q}{n}$$

*for some constant $q$. Assume also that given some $k \leq k_{\max} - 1$, for all $k' \in \{0, \ldots, k\}$,*

$$\left\| \zeta_{k'}^N \right\|_{L_{\mu^N}^2} \leq \sqrt{d}. \tag{F.24}$$

*Define*

$$\Theta(\boldsymbol{x}, \boldsymbol{x}') = \left\langle \boldsymbol{\alpha}_0^L(\mathbf{x}), \boldsymbol{\alpha}_0^L(\mathbf{x}') \right\rangle + \sum_{\ell=0}^{L-1} \left\langle \boldsymbol{\alpha}_0^\ell(\mathbf{x}), \boldsymbol{\alpha}_0^\ell(\mathbf{x}') \right\rangle \left\langle \boldsymbol{\beta}_0^\ell(\mathbf{x}), \boldsymbol{\beta}_0^\ell(\mathbf{x}') \right\rangle,$$

$$\tilde{\Delta}_k^N = \sup_{\substack{(\boldsymbol{x}, \boldsymbol{x}') \in \mathcal{M} \times \mathcal{M}, \\ k' \in \{0, \ldots, k\}}} \left| \Theta_{k'}^N(\boldsymbol{x}, \boldsymbol{x}') - \Theta(\boldsymbol{x}, \boldsymbol{x}') \right|.$$

*Assume $n \geq KL^{36+8q}d^9$, $d \geq K'd_0 \log(nn_0 C_{\mathcal{M}})$ for constants $K, K'$. Then on an event of probability at least $1 - e^{-cd}$*

$$\tilde{\Delta}_k^N \leq C \log^{3/4}(L) d^{3/4} L^{4+2q/3} n^{11/12}$$

*for some constants $c, C$.*

*Proof.* Recall that

$$\begin{aligned}
\Theta_k^N(\boldsymbol{x}, \boldsymbol{x}') &= \int_{s=0}^1 \left. \frac{\partial \tilde{f}_{\boldsymbol{\theta}}(\boldsymbol{x})}{\partial \boldsymbol{\theta}} \right|_{\boldsymbol{\theta}_{k+s}^N} \mathrm{d}s \left. \frac{\partial \tilde{f}_{\boldsymbol{\theta}}(\boldsymbol{x})}{\partial \boldsymbol{\theta}} \right|_{\boldsymbol{\theta}_k^N} \\
&= \sum_{\ell=0}^L \int_{s=0}^1 \left\langle \boldsymbol{\alpha}_{k+s}^\ell(\boldsymbol{x}), \boldsymbol{\alpha}_k^\ell(\boldsymbol{x}') \right\rangle \left\langle \boldsymbol{\beta}_{k+s}^\ell(\boldsymbol{x}), \boldsymbol{\beta}_k^\ell(\boldsymbol{x}') \right\rangle \mathrm{d}s
\end{aligned}$$

with the convention $\boldsymbol{\beta}_t^L(\boldsymbol{x}) = 1$ for all $t, \boldsymbol{x}$, and the parameters $\boldsymbol{\theta}_t^N$ given by (F.1). We thus have

$$\left| \Theta_k^N(\boldsymbol{x}, \boldsymbol{x}') - \Theta(\boldsymbol{x}, \boldsymbol{x}') \right| \leq \sum_{\ell=0}^L \int_{s=0}^1 \left| \begin{array}{c} \left\langle \boldsymbol{\alpha}_{k+s}^\ell(\boldsymbol{x}), \boldsymbol{\alpha}_k^\ell(\boldsymbol{x}') \right\rangle \left\langle \boldsymbol{\beta}_{k+s}^\ell(\boldsymbol{x}), \boldsymbol{\beta}_k^\ell(\boldsymbol{x}') \right\rangle \\ - \left\langle \boldsymbol{\alpha}_0^\ell(\boldsymbol{x}), \boldsymbol{\alpha}_0^\ell(\boldsymbol{x}') \right\rangle \left\langle \boldsymbol{\beta}_0^\ell(\boldsymbol{x}), \boldsymbol{\beta}_0^\ell(\boldsymbol{x}') \right\rangle \end{array} \right| \mathrm{d}s.$$

We consider a single summand in the above expression. On the event defined in lemma F.4, for all $\boldsymbol{x}, \boldsymbol{x}' \in \mathcal{M}, \ell \in \{0, \dots, L\}$,

$$\left|\langle \boldsymbol{\alpha}^\ell_{k+s}(\boldsymbol{x}), \boldsymbol{\alpha}^\ell_k(\boldsymbol{x}')\rangle \langle \boldsymbol{\beta}^\ell_{k+s}(\boldsymbol{x}), \boldsymbol{\beta}^\ell_k(\boldsymbol{x}')\rangle - \langle \boldsymbol{\alpha}^\ell_0(\boldsymbol{x}), \boldsymbol{\alpha}^\ell_0(\boldsymbol{x}')\rangle \langle \boldsymbol{\beta}^\ell_0(\boldsymbol{x}), \boldsymbol{\beta}^\ell_0(\boldsymbol{x}')\rangle\right|$$

$$\leq \left( \begin{array}{l} \left|\langle \boldsymbol{\alpha}^\ell_{k+s}(\boldsymbol{x}) - \boldsymbol{\alpha}^\ell_0(\boldsymbol{x}), \boldsymbol{\alpha}^\ell_k(\boldsymbol{x}')\rangle \langle \boldsymbol{\beta}^\ell_{k+s}(\boldsymbol{x}), \boldsymbol{\beta}^\ell_k(\boldsymbol{x}')\rangle\right| \\ + \left|\langle \boldsymbol{\alpha}^\ell_0(\boldsymbol{x}), \boldsymbol{\alpha}^\ell_k(\boldsymbol{x}') - \boldsymbol{\alpha}^\ell_0(\boldsymbol{x}')\rangle \langle \boldsymbol{\beta}^\ell_{k+s}(\boldsymbol{x}), \boldsymbol{\beta}^\ell_k(\boldsymbol{x}')\rangle\right| \\ + \left|\langle \boldsymbol{\alpha}^\ell_0(\boldsymbol{x}), \boldsymbol{\alpha}^\ell_0(\boldsymbol{x}')\rangle \langle \boldsymbol{\beta}^\ell_{k+s}(\boldsymbol{x}) - \boldsymbol{\beta}^\ell_0(\boldsymbol{x}), \boldsymbol{\beta}^\ell_k(\boldsymbol{x}')\rangle\right| \\ + \left|\langle \boldsymbol{\alpha}^\ell_0(\boldsymbol{x}), \boldsymbol{\alpha}^\ell_0(\boldsymbol{x}')\rangle \langle \boldsymbol{\beta}^\ell_0(\boldsymbol{x}), \boldsymbol{\beta}^\ell_k(\boldsymbol{x}') - \boldsymbol{\beta}^\ell_0(\boldsymbol{x}')\rangle\right| \end{array} \right)$$

$$\leq C \left( \begin{array}{l} n\left\|\boldsymbol{\alpha}^\ell_{k+s}(\boldsymbol{x}) - \boldsymbol{\alpha}^\ell_0(\boldsymbol{x})\right\|_2 + n\left\|\boldsymbol{\alpha}^\ell_k(\boldsymbol{x}') - \boldsymbol{\alpha}^\ell_0(\boldsymbol{x}')\right\|_2 \\ + \sqrt{n}\left\|\boldsymbol{\beta}^\ell_{k+s}(\boldsymbol{x}) - \boldsymbol{\beta}^\ell_0(\boldsymbol{x})\right\|_2 + \sqrt{n}\left\|\boldsymbol{\beta}^\ell_k(\boldsymbol{x}') - \boldsymbol{\beta}^\ell_0(\boldsymbol{x}')\right\|_2 \end{array} \right)$$

$$\leq C' \left( L^{3/2+q}\sqrt{dn} + \log^{3/4}(L)d^{3/4}L^{3+2q/3}n^{11/12} \right)$$

$$\leq C'' \log^{3/4}(L)d^{3/4}L^{3+2q/3}n^{11/12},$$

for some constants. Summing this bound over $\ell$ gives the desired result. $\qquad\square$

## F.5 AUXILIARY LEMMAS AND PROOFS

**Lemma F.6.** *Consider a collection of $n$ i.i.d. variables $X_i = g_i^2, g_i \sim \mathcal{N}(0, \frac{1}{n})$ and denote the order statistics by $X_{(i)}$ (so that $X_{(1)} \leq X_{(2)} \dots$). For any $d \geq K' \log n$, $n \geq K'' d^3$, $\eta > C\frac{d^{9/8}}{n^{3/4}}$ and integer $Kn\eta^{2/3} \leq k \leq n$, where $K, K', K''$ are appropriately chosen absolute constants, we have*

$$\mathbb{P}\left[\sum_{i=1}^k X_{(i)} \geq \eta^2\right] \geq 1 - C'e^{-cd},$$

*where $c, C, C'$ are absolute constants.*

*Proof.* We will relate sums of order statistics of $X_i$ to functions of uniform order statistics and show that these concentrate. We denote the CDF of the $X_i$ and its inverse by $F$ and $F^{-1}$ respectively. We use

$$(X_{(1)}, ..., X_{(k)}) \overset{d}{=} (F^{-1}(U_{(1)}), ..., F^{-1}(U_{(k)}))$$

where $U_{(i)}$ are order statistics with respect to $\text{Unif}(0, 1)$ (David, 2011). Since $X_i \sim \frac{1}{n}Y_i, Y_i \sim \chi_1^2$ we have

$$F(x) = \text{erf}(\sqrt{\frac{nx}{2}})$$

$$F^{-1}(t) = \frac{2}{n}(\text{erf}^{-1}(t))^2 \geq \frac{c_0}{n}t^2$$

where in the inequality we used the series representation of $\text{erf}^{-1}$. This gives

$$\sum_{i=1}^k X_{(i)} \overset{d}{=} \sum_{i=1}^k F^{-1}(U_{(i)}) \geq \frac{c_0}{n}\sum_{i=1}^k U_{(i)}^2.$$

The joint PDF of the first $k$ order statistics for any distribution admitting a density is given by

$$f_{(1)...(k)}(x_1, ..., x_k) = \frac{n!}{(n-k)!}(1 - F(x_k))^{n-k}\prod_{i=1}^k f(x_i)$$

where $x_1 \leq x_2 \leq ... \leq x_k$ (David, 2011). Applying this to the uniform order statistics, we can compute the mean of the summands

$$\mathbb{E}U_{(i)}^2$$

$$= \frac{n!}{(n-k)!}\int\limits_0^{u_2}...\int\limits_0^{u_k}\int\limits_0^1 u_i^2\left(1-u_k\right)^{n-k}du_k...du_1 = \frac{n!i(i+1)}{(n-k)!(k+1)!}\int\limits_0^1 u_k^{k+1}\left(1-u_k\right)^{n-k}du_k$$

$$=\frac{i(i+1)}{(n+2)(n+1)}$$

$$\mathbb{E}\sum_{i=1}^k U_{(i)}^2 = \frac{k(k+1)(k+2)}{3(n+2)(n+1)} \geq \frac{c_1 k^3}{n^2}.$$

In order to show concentration, we appeal to the Rényi representation of order statistics (Boucheron et al., 2012). This allows us to write $\sum_{i=1}^k U_{(i)}^2$ as a Lipschitz function of *independent* exponential random variables, and we can then apply standard concentration results for such functions (Talagrand, 1995). This representation is due to a useful property unique to the exponential distribution whereby the differences between order statistics are independent exponentially distributed variables themselves when properly normalized.

If we define by $E_i, ..., E_n$ a collection of independent standard exponential variables, the Rényi representation of the uniform order statistics gives

$$(U_{(1)}, ..., U_{(k)}) \overset{d}{=} \left(1 - \exp\left(-\frac{E_1}{n}\right), ..., 1 - \exp\left(-\sum_{j=1}^k \frac{E_j}{n-j+1}\right)\right).$$

We now truncate the $(E_1, ..., E_k)$, so that w.p. $\mathbb{P} \geq 1 - ke^{-K}$ we have $\forall i : E_i \in [0, K]$, and denote this event by $\mathcal{E}_K$. Using $K < \frac{n}{2}$, it is evident that $\sum_{i=1}^k U_{(i)}^2$ is equal in distribution to a convex function of $(E_1, ..., E_k)$ after truncation (which can be seen by calculating second derivatives). The Lipschitz constant of this function is bounded by $\frac{4k}{n-k} \doteq \lambda$.

If we define rescaled variables $\tilde{E}_i = \lambda E_i$ then with the same probability they take values in $[0, K\lambda]$. $\sum_{i=1}^k U_{(i)}^2$ written in terms of $\tilde{E}_i$ is now 1-Lipschitz and convex, and we can apply Talagrand's concentration inequality (Talagrand, 1995) to obtain

$$\mathbb{P}\left[\left|\sum_{i=1}^k \mathbb{1}_{\mathcal{E}_K}U_{(i)}^2 - \mathbb{E}\mathbb{1}_{\mathcal{E}_K}\sum_{i=1}^k U_{(i)}^2\right| \geq tK\lambda\right] \leq C\exp\left(-ct^2\right).$$

Setting $t = \frac{c_1 k^3}{2n^2 K\lambda} = \frac{c_1 k^2(n-k)}{8n^2 K}$, if we now assume

$$c_2 n\eta^{2/3} \leq k \leq c'n$$

for some $c' < 1$ we obtain

$$\mathbb{P}\left[\left|\sum_{i=1}^k \mathbb{1}_{\mathcal{E}_K}U_{(i)}^2 - \mathbb{E}\sum_{i=1}^k \mathbb{1}_{\mathcal{E}_K}U_{(i)}^2\right| \geq \frac{c_1 k^3}{2n^2}\right] \leq C\exp\left(-\frac{c''k^4}{n^2 K^2}\right) \leq C\exp\left(-\frac{c''n^2\eta^{8/3}}{K^2}\right).$$

We would also like to ensure that the truncation does not cause a large deviation in the mean. We have

$$\underset{\{U_{(i)}\}}{\mathbb{E}}\sum_{i=1}^k U_{(i)}^2 - \mathbb{1}_{\mathcal{E}_K}\sum_{i=1}^k U_{(i)}^2$$

$$= \underset{\{E_i\}}{\mathbb{E}}\sum_{i=1}^k \left(1 - \exp\left(-\sum_{j=1}^i \frac{E_j}{n-i+1}\right)\right)^2 - \mathbb{1}_{\mathcal{E}_K}\sum_{i=1}^k \left(1 - \exp\left(-\sum_{j=1}^i \frac{E_j}{n-i+1}\right)\right)^2$$

$$\leq \sum_{m=1}^{l} \underset{\{E_i\}}{\mathbb{E}} \mathbb{1}_{E_m > k} \sum_{i=1}^{k} \left(1 - \exp\left(-\sum_{j=1}^{i} \frac{E_j}{n-i+1}\right)\right)^2 \leq k \sum_{m=1}^{l} \underset{\{E_i\}}{\mathbb{E}} \mathbb{1}_{E_m > K} = k^2 \underset{\{E_i\}}{\mathbb{E}} \mathbb{1}_{E_1 > K}$$

$$= k^2 e^{-K}.$$

Since we would like this to be small compared to $\mathbb{E} \sum_{i=1}^{k} U_{(i)}^2 \geq \frac{c_1 k^3}{n^2}$ we can require $K > \log \frac{4n^2}{c_1 k}$

which gives $\underset{\{U_{(i)}\}}{\mathbb{E}} \sum_{i=1}^{k} U_{(i)}^2 - \mathbb{1}_{\mathcal{E}_K} \sum_{i=1}^{k} U_{(i)}^2 < \frac{c_1 k^3}{4n^2}$. We can then choose the constant $c_2$ such that with

probability $\mathbb{P} \geq 1 - \exp\left(\log k - K\right) - C \exp\left(-\frac{cn^2 \eta^{8/3}}{K^2}\right)$

$$\sum_{i=1}^{k} X_{(i)} \geq \frac{c_0}{n} \mathbb{1}_{\mathcal{E}_K} \sum_{i=1}^{k} U_{(i)}^2 \geq \frac{c_0}{n} \left(\mathbb{E}\mathbb{1}_{\mathcal{E}_K} \sum_{i=1}^{k} U_{(i)}^2 - \frac{c_1 k^3}{2n^2}\right)$$

$$= \frac{c_0}{n} \left(\mathbb{E} \sum_{i=1}^{k} U_{(i)}^2 - \frac{c_1 k^3}{2n^2} + \mathbb{E}\mathbb{1}_{\mathcal{E}_K} \sum_{i=1}^{k} U_{(i)}^2 - \mathbb{E} \sum_{i=1}^{k} U_{(i)}^2\right)$$

$$\geq \frac{c_0 c_1 k^3}{4n^2} \geq \eta^2.$$

The upper bound on $k$ can then be removed since the inequality then applies to all larger $k$ automatically. If we now set $\eta$ according to equation (F.2), and choose $K = d \geq K' \log n$ and $n$ satisfying $n \geq K'' d^3$ for appropriate constants $K', K''$ we obtain

$$\mathbb{P}\left[\sum_{i=1}^{k} X_{(i)} \geq \eta^2\right] \geq 1 - \exp\left(\log k - d\right) - C \exp\left(-\frac{c C_\eta^{8/3} L^{4+8q/3} n^{2/3}}{d^2}\right) \geq 1 - C' e^{-c'' d},$$

and due to our choice of $\eta$, this result holds for all $k \geq C n \eta^{2/3} \geq C C_\eta^{2/3} n^{2/3} L^{1+2q/3}$. $\qquad \square$

*Proof of lemma F.2.* i) We begin by controlling the pre-activation norms. Considering a point $\overline{\boldsymbol{x}} \in N_{n^{-3} n_0^{-1/2}}$, where $N_{n^{-3} n_0^{-1/2}}$ is the net defined in Appendix D.3.1, rotational invariance of the Gaussian distribution gives

$$\left\|\boldsymbol{\rho}_0^\ell(\overline{\boldsymbol{x}})\right\|_2^2 = \boldsymbol{\alpha}_0^{\ell-1*}(\overline{\boldsymbol{x}}) \boldsymbol{W}_0^{\ell*} \boldsymbol{W}_0^\ell \boldsymbol{\alpha}_0^{\ell-1}(\overline{\boldsymbol{x}}) \overset{d}{=} \left\|\boldsymbol{\alpha}_0^{\ell-1}(\overline{\boldsymbol{x}})\right\|_2^2 \left\|\left(\boldsymbol{W}_0^\ell\right)_{(:,1)}\right\|_2^2$$

where $\left(\boldsymbol{W}_0^\ell\right)_{(:,1)}$ is the first column of $\boldsymbol{W}_0^\ell$. Bernstein's inequality then gives

$$\mathbb{P}\left[\left\|\boldsymbol{\rho}_0^\ell(\overline{\boldsymbol{x}})\right\|_2^2 \leq C \left\|\boldsymbol{\alpha}_0^{\ell-1}(\overline{\boldsymbol{x}})\right\|_2^2\right] \geq 1 - C' e^{-cn}$$

for appropriate constants. As discussed in Lemma D.8, if we choose $d$ to satisfy the requirements of this lemma then $\left|N_{n^{-3} n_0^{-1/2}}\right| \leq e^{C'' d}$ for some constant. We can then uniformize over the net using a union bound, obtaining

$$\mathbb{P}\left[\forall \overline{\boldsymbol{x}} \in N_{n^{-3} n_0^{-1/2}} : \quad \left\|\boldsymbol{\rho}_0^\ell(\overline{\boldsymbol{x}})\right\|_2^2 \leq C \left\|\boldsymbol{\alpha}_0^{\ell-1}(\overline{\boldsymbol{x}})\right\|_2^2\right] \geq 1 - C' e^{C'' d - cn} \geq 1 - C' e^{-c' n}$$

for some $c'$, assuming $n \geq Kd$. We now need to control the feature norms and pre-activation norms off of the net. From (D.62) and lemma G.10 we obtain that for $d$ satisfying the requirements of lemma D.9,

$$\mathbb{P}\left[\begin{array}{c} \forall \boldsymbol{x} \in \mathcal{M}, \ell \in [L] : \exists \overline{\boldsymbol{x}} \in N_{n^{-3} n_0^{-1/2}} \cap \mathcal{N}_{n^{-3} n_0^{-1/2}}(\boldsymbol{x}) \\ s.t. \\ \left\{\left\|\boldsymbol{\rho}_0^\ell(\boldsymbol{x}) - \boldsymbol{\rho}_0^\ell(\overline{\boldsymbol{x}})\right\|_2 \leq C n^{-5/2}\right\} \cap \left\{\left|\left\|\boldsymbol{\alpha}_0^{\ell-1}(\overline{\boldsymbol{x}})\right\|_2 - 1\right| \leq \frac{1}{2}\right\} \end{array}\right] \geq 1 - e^{-cd}.$$

By taking a union bound over the above two results, we obtain

$$\mathbb{P}\left[\forall \boldsymbol{x} \in \mathcal{M}, \ell \in [L] : \quad \left\|\boldsymbol{\rho}_0^\ell(\boldsymbol{x})\right\|_2 \leq C\right] \geq 1 - C' e^{-c' d} \tag{F.25}$$

for some constants

We next turn to controlling the generalized backward features and transfer matrices. Our first task is to bound the number of support patterns that can be encountered, namely $\left|\overline{\mathcal{J}}_\eta(\mathcal{M})\right|$. In order to do this it will be convenient to introduce a set that contains $\overline{\mathcal{J}}_\eta(\mathcal{M})$ and is easier to reason about. We define a metric between supports by

$$d_{\mathrm{supp}}(S, S') = |S \ominus S'|$$

and denote by $\mathbb{B}_s(S, \delta) \subset \mathcal{P}([n])$ a ball defined with respect to this metric, where $\delta \in \{0\} \cup [n]$ and $\mathcal{P}(A)$ is the power set of a set $A$. For $\delta_\eta(\boldsymbol{y})$ and $\mathcal{B}(\boldsymbol{y}, \eta)$ defined in (F.8) and (F.4) respectively, it is clear that

$$\mathcal{B}(\boldsymbol{y}, \eta) \subseteq \mathbb{B}_s(\mathrm{supp}(\boldsymbol{y} > \boldsymbol{0}), \delta_\eta(\boldsymbol{y}))$$

and consequently

$$\overline{\mathcal{J}}_\eta(\mathcal{M}) \subseteq \bigcup_{\boldsymbol{x} \in \mathcal{M}} \overset{L}{\underset{\ell=1}{\otimes}} \mathbb{B}_s(\mathrm{supp}(\boldsymbol{\rho}_0^\ell(\boldsymbol{x}) > \boldsymbol{0}), \delta_\eta(\boldsymbol{\rho}_0^\ell(\boldsymbol{x}))). \tag{F.26}$$

We will aim to control the volume of this set, which we will achieve by controlling it first on a net. This will require transferring control between different nearby points.

For any $S, S' \in [n]$ and $\delta \in \{0\} \cup [n]$, the triangle inequality implies

$$\mathbb{B}_s(S, \delta) \subseteq \mathbb{B}_s(S', \delta + d_s(S, S')).$$

For some $\boldsymbol{p} \in \mathbb{B}_E(\boldsymbol{g}, r)$, we also have

$$\begin{aligned}
\delta_\eta(\boldsymbol{p}) &= \max_{\boldsymbol{y} \in \mathbb{B}_E(\boldsymbol{p}, \eta)} d_s(\boldsymbol{p}, \boldsymbol{y}) \leq d_s(\boldsymbol{p}, \boldsymbol{g}) + \max_{\boldsymbol{y} \in \mathbb{B}_E(\boldsymbol{p}, \eta)} d_s(\boldsymbol{g}, \boldsymbol{y}) \\
&\leq d_s(\boldsymbol{p}, \boldsymbol{g}) + \max_{\boldsymbol{y} \in \mathbb{B}_E(\boldsymbol{g}, \eta+r)} d_s(\boldsymbol{g}, \boldsymbol{y}) \\
&= d_s(\boldsymbol{p}, \boldsymbol{g}) + \delta_{\eta+r}(\boldsymbol{g})
\end{aligned} \tag{F.27}$$

where we used $\mathbb{B}_E(\boldsymbol{p}, \eta) \subseteq \mathbb{B}_E(\boldsymbol{g}, \eta + r)$. It follows that

$$\begin{aligned}
\mathbb{B}_s(\mathrm{supp}(\boldsymbol{p} > \boldsymbol{0}), \delta_\eta(\boldsymbol{p})) &\subseteq \mathbb{B}_s(\mathrm{supp}(\boldsymbol{g} > \boldsymbol{0}), \delta_\eta(\boldsymbol{p}) + d_s(\boldsymbol{p}, \boldsymbol{g})) \\
&\subseteq \mathbb{B}_s(\mathrm{supp}(\boldsymbol{g} > \boldsymbol{0}), \delta_{\eta+r}(\boldsymbol{g}) + 2d_s(\boldsymbol{p}, \boldsymbol{g})).
\end{aligned} \tag{F.28}$$

From (D.62) and lemma G.10 we obtain that for $d$ satisfying the requirements of lemma D.8,

$$\mathbb{P}\left[\begin{array}{c} \forall \boldsymbol{x} \in \mathcal{M}, \ell \in [L] : \exists \overline{\boldsymbol{x}} \in N_{n^{-3}n_0^{-1/2}} \cap \mathcal{N}_{n^{-3}n_0^{-1/2}}(\boldsymbol{x}) \\ s.t. \\ \left\{\left\|\boldsymbol{\rho}_0^\ell(\boldsymbol{x}) - \boldsymbol{\rho}_0^\ell(\overline{\boldsymbol{x}})\right\|_2 \leq Cn^{-5/2}\right\} \cap \left\{d_s(\boldsymbol{\rho}_0^\ell(\boldsymbol{x}), \boldsymbol{\rho}_0^\ell(\overline{\boldsymbol{x}})) \leq d\right\} \end{array}\right] \geq 1 - 6e^{-d/2}, \tag{F.29}$$

since under the assumptions of the lemma $d_s(\boldsymbol{\rho}_0^\ell(\boldsymbol{x}), \boldsymbol{\rho}_0^\ell(\overline{\boldsymbol{x}})) \leq \sum_{\ell=1}^{L} \left|R_\ell\left(\overline{\boldsymbol{x}}, Cn^{-3}\right)\right|$, with $R_\ell\left(\overline{\boldsymbol{x}}, Cn^{-3}\right)$ denoting the number of risky features as defined in section D.3.1. We denote this event by $\mathcal{E}_\rho$.

On $\mathcal{E}_\rho$, we can transfer control of the ball of feature supports from a point on the net to any point on the manifold. For some $\ell, \boldsymbol{x}$ we denote by $\overline{\boldsymbol{x}}$ the point on the net that satisfies the above condition. Considering (F.28), we choose $\boldsymbol{g} = \boldsymbol{\rho}_0^\ell(\overline{\boldsymbol{x}})$, $\boldsymbol{p} = \boldsymbol{\rho}_0^\ell(\boldsymbol{x})$, $r = Cn^{-5/2}$ and $\eta = C_\eta L^{3/2+q}n^{-1/2}$, obtaining

$$\begin{aligned}
&\mathbb{B}_s(\mathrm{supp}(\boldsymbol{\rho}_0^\ell(\boldsymbol{x}) > \boldsymbol{0}), \delta_\eta(\boldsymbol{\rho}_0^\ell(\boldsymbol{x}))) \\
\subseteq\ & \mathbb{B}_s(\mathrm{supp}(\boldsymbol{\rho}_0^\ell(\overline{\boldsymbol{x}}) > \boldsymbol{0}), \delta_{\eta+Cn^{-5/2}}(\boldsymbol{\rho}_0^\ell(\overline{\boldsymbol{x}})) + 2d_s(\boldsymbol{\rho}_0^\ell(\boldsymbol{x}), \boldsymbol{\rho}_0^\ell(\overline{\boldsymbol{x}}))) \\
\subseteq\ & \mathbb{B}_s(\mathrm{supp}(\boldsymbol{\rho}_0^\ell(\overline{\boldsymbol{x}}) > \boldsymbol{0}), \delta_{2\eta}(\boldsymbol{\rho}_0^\ell(\overline{\boldsymbol{x}})) + 2d),
\end{aligned} \tag{F.30}$$

where we assumed $C_\eta L^{3/2+q} n^2 > C$.

We next turn to controlling $\delta_{2\eta}(\boldsymbol{\rho}_0^\ell(\overline{\boldsymbol{x}}))$, which is now a random variable, for all $\ell \in [L], \overline{\boldsymbol{x}} \in N_{n^{-3}n_0^{-1/2}}$. From lemma F.1 we have for a vector $\boldsymbol{g}$ with $g_i \sim_{\mathrm{iid}} \mathcal{N}(0, \frac{1}{n})$,

$$\mathbb{P}\left[\delta_{2\eta}(\boldsymbol{g}) \geq C_0 n\eta^{2/3}\right] \leq C' e^{-cd}.$$

Considering a vector $\boldsymbol{\rho}_0^\ell(\overline{\boldsymbol{x}})$ for some $\ell \in [L], \overline{\boldsymbol{x}} \in N_{n^{-3}n_0^{-1/2}}$, we have

$$\boldsymbol{\rho}_0^\ell(\overline{\boldsymbol{x}}) \sim \mathcal{N}(0, 2 \left\| \boldsymbol{\alpha}_0^{\ell-1}(\overline{\boldsymbol{x}}) \right\|_2^2 n^{-1}).$$

Lemma D.2 then gives

$$\mathbb{P} \left[ \sqrt{2} \left\| \boldsymbol{\alpha}_0^{\ell-1}(\overline{\boldsymbol{x}}) \right\|_2 < 1 \right] \leq C\ell e^{-cd} \leq C e^{-c'd}$$

for some constants, assuming $d > K \log L$ for some $K$. Since on the complement of this event we have $\sqrt{2}\eta \left\| \boldsymbol{\alpha}_0^{\ell-1}(\overline{\boldsymbol{x}}) \right\|_2^{-1} \leq 2\eta$, lemma F.1 and a rescaling gives

$$\begin{aligned}
\mathbb{P} \left[ \delta_{2\eta}(\boldsymbol{\rho}_0^\ell(\overline{\boldsymbol{x}})) \geq C_0 n\eta^{2/3} \right] &= \mathbb{P} \left[ \delta_{\sqrt{2}\eta\|\boldsymbol{\alpha}^{\ell-1}(\overline{\boldsymbol{x}})\|_2^{-1}}(\boldsymbol{g}) \geq C_0 n\eta^{2/3} \right] \\
&\leq \mathbb{P} \left[ \delta_{2\eta}(\boldsymbol{g}) \geq C_0 n\eta^{2/3} \right] + \mathbb{P} \left[ \sqrt{2} \left\| \boldsymbol{\alpha}^{\ell-1}(\overline{\boldsymbol{x}}) \right\|_2 < 1 \right] \quad \text{(F.31)} \\
&\leq Ce^{-cd} + C'e^{-c'd} \leq C''e^{-c''d}.
\end{aligned}$$

for some constants. Taking a union bound over $N_{n^{-3}n_0^{-1/2}}$ and $[L]$ we obtain

$$\mathbb{P} \left[ \exists \overline{\boldsymbol{x}} \in N_{n^{-3}n_0^{-1/2}}, \ell \in [L] \ s.t. \ \delta_{2\eta}(\boldsymbol{\rho}_0^\ell(\overline{\boldsymbol{x}})) \geq C_0 n\eta^{2/3} \right] \leq \left| N_{n^{-3}n_0^{-1/2}} \right| LCe^{-cd} \leq C'e^{-c'd} \tag{F.32}$$

under the same assumptions on $d$ as in lemma D.8, and additionally assuming $d \geq K \log L$ for some $K$.

Since $n, d, \eta$ satisfy the assumptions of lemma F.1, we have $n\eta^{2/3} \geq C'n^{1/2}d^{3/4} \geq C'd$ for some $C'$ and hence

$$\mathbb{P} \left[ \exists \overline{\boldsymbol{x}} \in N_{n^{-3}n_0^{-1/2}}, \ell \in [L] \ s.t. \ \delta_{2\eta}(\boldsymbol{\rho}_0^\ell(\overline{\boldsymbol{x}})) + 2d \geq C_1 n\eta^{2/3} \right] \leq Ce^{-cd} \tag{F.33}$$

for some constants $c, C, C_1$. Denoting the complement of above event by $\mathcal{E}_\delta^N$, we find that on $\mathcal{E}_\rho \cap \mathcal{E}_\delta^N$, for every $\boldsymbol{x}$ we can find $\overline{\boldsymbol{x}} \in N_{n^{-3}n_0^{-1/2}} \cap N_{n^{-3}n_0^{-1/2}}(\boldsymbol{x})$ such that

$$\begin{aligned}
\mathbb{B}_s(\text{supp}(\boldsymbol{\rho}_0^\ell(\boldsymbol{x}) > \boldsymbol{0}), \delta_\eta(\boldsymbol{\rho}_0^\ell(\boldsymbol{x}))) &\subseteq \mathbb{B}_s(\text{supp}(\boldsymbol{\rho}_0^\ell(\overline{\boldsymbol{x}}) > \boldsymbol{0}), \delta_{2\eta}(\boldsymbol{\rho}_0^\ell(\overline{\boldsymbol{x}})) + 2d) \\
&\subseteq \mathbb{B}_s(\text{supp}(\boldsymbol{\rho}_0^\ell(\overline{\boldsymbol{x}}) > \boldsymbol{0}), C_1 n\eta^{2/3}),
\end{aligned}$$

where we used (F.30). On $\mathcal{E}_\rho \cap \mathcal{E}_\delta^N$, we can thus bound the size of the set that contains $\overline{\mathcal{J}}_\eta$, denoting its size by

$$S_\eta = \text{Vol} \bigcup_{\boldsymbol{x} \in \mathcal{M}} \overset{L}{\underset{\ell=1}{\otimes}} \mathbb{B}_s(\text{sign}(\boldsymbol{\rho}_0^\ell(\boldsymbol{x})), \delta_\eta(\boldsymbol{\rho}_0^\ell(\boldsymbol{x}))).$$

We first note that for any $\boldsymbol{p}$,

$$\text{Vol}\mathbb{B}_s(\boldsymbol{p}, C_1 n\eta^{2/3}) \leq \sum_{i=0}^{\lceil C_1 n\eta^{2/3} \rceil} \binom{n}{i} \leq C \lceil C_1 n\eta^{2/3} \rceil n^{\lceil C_1 n\eta^{2/3} \rceil} \leq C'n^{C''C_1 n\eta^{2/3}}$$

for appropriate constants, assuming $n\eta^{2/3} > K \log(n\eta^{2/3})$ for some $K$. It follows that on $\mathcal{E}_\rho \cap \mathcal{E}_\delta^N$,

$$\begin{aligned}
S_\eta &= \text{Vol} \bigcup_{\boldsymbol{x} \in \mathcal{M}} \overset{L}{\underset{\ell=1}{\otimes}} \mathbb{B}_s(\text{supp}(\boldsymbol{\rho}_0^\ell(\boldsymbol{x})), \delta_\eta(\boldsymbol{\rho}^\ell(\boldsymbol{x}))) \leq \text{Vol} \bigcup_{\boldsymbol{x} \in \mathcal{M}} \overset{L}{\underset{\ell=1}{\otimes}} \mathbb{B}_s(\text{supp}(\boldsymbol{\rho}_0^\ell(\overline{\boldsymbol{x}})), C_1 n\eta^{2/3}) \\
&\leq \prod_{\ell=1}^L \sum_{\overline{\boldsymbol{x}} \in N_{n^{-3}n_0^{-1/2}}} \text{Vol}\mathbb{B}_s(\text{supp}(\boldsymbol{\rho}_0^\ell(\overline{\boldsymbol{x}})), C_1 n\eta^{2/3}) \\
&\leq C' \left| N_{n^{-3}n_0^{-1/2}} \right| e^{CdLn\eta^{2/3}} \leq C'e^{C''dLn\eta^{2/3}}
\end{aligned}$$

for appropriate constants, since $n\eta^{2/3} \geq C'''d$ and $d$ satisfies the assumptions of lemma D.8. Since after worsening constants we have $\mathbb{P} \left[ \overline{\mathcal{E}_\rho \cap \mathcal{E}_\delta^N} \right] \leq C'e^{-cd}$, we obtain

$$\mathbb{P} \left[ S_\eta > C'e^{C''dLn\eta^{2/3}} \right] \leq C'e^{-cd} \tag{F.34}$$

for some constants.

We would next like to employ lemma D.14 in order to control the quantities of interest for a single $\mathcal{J} \in \overline{\mathcal{J}}_\eta(\mathcal{M})$, and then take a union bound utilizing the upper bound above on $|\overline{\mathcal{J}}_\eta|$. This will require controlling the event $\mathcal{E}_{\delta K}$ in the lemma statement with an appropriate choice of the constants $\delta_s, K_s$. As in other sections, we use the convention $\boldsymbol{\Gamma}_{\mathcal{J}0}^{\ell:\ell+1} = \boldsymbol{I}$ for any $\ell \in [L]$.

At a given collection of supports $\mathcal{J} \in \mathcal{J}_\eta(\boldsymbol{x})$ for some $\boldsymbol{x} \in \mathcal{M}$, we choose $\boldsymbol{x}$ as the anchor point in lemma D.14.

From (F.27) we have, for any $\overline{\boldsymbol{x}} \in N_{n^{-3}n_0^{-1/2}}$,

$$\delta_\eta(\boldsymbol{\rho}_0^\ell(\boldsymbol{x})) \le d_s\left(\boldsymbol{\rho}_0^\ell(\boldsymbol{x}), \boldsymbol{\rho}_0^\ell(\overline{\boldsymbol{x}})\right) + \delta_{\eta + \left\|\boldsymbol{\rho}_0^\ell(\boldsymbol{x}) - \boldsymbol{\rho}_0^\ell(\overline{\boldsymbol{x}})\right\|_2}(\boldsymbol{\rho}_0^\ell(\overline{\boldsymbol{x}})).$$

Then using (F.29) we obtain to bound the two terms in the RHS gives

$$\mathbb{P}\left[\forall \boldsymbol{x} \in \mathcal{M}, \ell \in [L] : \exists \overline{\boldsymbol{x}} \in N_{n^{-3}n_0^{-1/2}} \cap N_{n^{-3}n_0^{-1/2}}(\boldsymbol{x}) \ \ s.t. \ \ \delta_\eta(\boldsymbol{\rho}_0^\ell(\boldsymbol{x})) \le d + \delta_{2\eta}(\boldsymbol{\rho}_0^\ell(\overline{\boldsymbol{x}}))\right]$$
$$\ge 1 - 6e^{-d/2}.$$

where we used $\eta = C_\eta L^{3/2+q} n^{-1/2}$ and $d$ satisfies the requirements of lemma D.8. Using (F.33) to bound $d + \delta_{2\eta}(\boldsymbol{\rho}_0^\ell(\overline{\boldsymbol{x}}))$ uniformly on $N_{n^{-3}n_0^{-1/2}}$ and $\ell$, and combining the failure probabilities of these events by a union bound, we obtain

$$\mathbb{P}\left[\exists \boldsymbol{x} \in \mathcal{M} \ \ s.t. \ \ \delta_\eta(\boldsymbol{\rho}_0^\ell(\boldsymbol{x})) > C_1 n \eta^{2/3}\right] \le 6e^{-d/2} + Ce^{-cd} \le C'e^{-c'd}$$

for some constants. Since $\delta_\eta(\boldsymbol{\rho}_0^\ell(\boldsymbol{x})) \ge |J_\ell \ominus I_{\ell,0}(\boldsymbol{x})|$ for any $J_\ell \in \mathcal{J} \in \mathcal{J}_\eta(\boldsymbol{x})$, implies directly that

$$\mathbb{P}\left[\forall \boldsymbol{x} \in \mathcal{M}, J_\ell \in \mathcal{J} \in \mathcal{J}_\eta(\boldsymbol{x}) : |J_\ell \ominus I_{\ell,0}(\boldsymbol{x})| \le C_1 n \eta^{2/3}\right] \ge 1 - C'e^{-c'd}. \tag{F.35}$$

In the notation of lemma D.14 we denote this event by $\mathcal{E}_\delta$, and choose $\delta_s = C_1 n \eta^{2/3}$.

From the definition of $\overline{\mathcal{J}}_\eta$, for every $\boldsymbol{x} \in \mathcal{M}$ and $J_\ell$ that is an element of $\mathcal{J} \in \overline{\mathcal{J}}_\eta(\boldsymbol{x})$,

$$J_\ell = \text{supp}\left(\boldsymbol{\rho}_0^\ell(\boldsymbol{x}) + \boldsymbol{v} > \boldsymbol{0}\right)$$

for some $\boldsymbol{v}$ such that $\|\boldsymbol{v}\|_2 \le \eta$. We now consider the vector

$$\boldsymbol{w} = \left(\boldsymbol{P}_{J_\ell} - \boldsymbol{P}_{I_\ell(\boldsymbol{x})}\right) \boldsymbol{\rho}_0^\ell(\boldsymbol{x}).$$

Note that for any element of $w_i$ that is non-zero, we must have $|v_i| \ge \left|\rho_0^\ell(\boldsymbol{x})_i\right|$ (since the perturbation must change the sign of this element), in which case we have $|w_i| = \left|\rho_0^\ell(\boldsymbol{x})_i\right|$. Denoting the set of indices of these non-zero elements by $Q$, we have

$$\|\boldsymbol{w}\|_2^2 = \sum_{i \in Q} w_i^2 = \sum_{i \in Q}\left(\rho_0^\ell(\boldsymbol{x})_i\right)^2 \le \sum_{i \in Q} v_i^2 \le \|\boldsymbol{v}\|_2^2 \le \eta^2.$$

This holds for all $\ell \in [L]$. Thus if we set $K_s = \eta$ for $K_s$, the event $\mathcal{E}_K$ in lemma D.14 holds with probability 1. We therefore choose

$$\mathcal{E}_{\delta K} = \mathcal{E}_\delta$$

with $\mathcal{E}_\delta$ defined in (F.35). In order to apply D.14 we must also ensure

$$\delta_s = C_0 n \eta^{2/3} \le \frac{n}{L}, \quad K_s = \eta \le \frac{1}{2}L^{-3/2}.$$

Setting $\eta = \frac{C_\eta L^{3/2+q}}{\sqrt{n}}$ as per (F.2), we can satisfy these requirements by demanding $n \ge C_0^3 C_\eta^2 L^{6+2q}$.

We are now in a position to apply lemma D.14 to control the objects of interest. We use rotational invariance of the Gaussian distribution repeatedly to obtain

$$\mathbb{1}_{\mathcal{E}_{\delta K}} \left\|\boldsymbol{\beta}_{\mathcal{J},0}^\ell\right\|_2 \quad = \mathbb{1}_{\mathcal{E}_{\delta K}} \left\|\boldsymbol{W}_0^{L+1} \boldsymbol{\Gamma}_{\mathcal{J}0}^{L:\ell+2} \boldsymbol{P}_{J_{\ell+1}}\right\|_2 \underset{a.s.}{\le} \mathbb{1}_{\mathcal{E}_{\delta K}} \left\|\boldsymbol{W}_0^{L+1} \boldsymbol{\Gamma}_{\mathcal{J}0}^{L:\ell+2}\right\|_2$$

$$\overset{d}{=} \mathbb{1}_{\mathcal{E}_{\delta K}} \left\|\boldsymbol{\Gamma}_{\mathcal{J}0}^{L:\ell+2} \boldsymbol{W}_0^{L+1*}\right\|_2 \overset{d}{=} \mathbb{1}_{\mathcal{E}_{\delta K}} \left\|\boldsymbol{\Gamma}_{\mathcal{J}0}^{L:\ell+2} \boldsymbol{e}_1\right\|_2 \left\|\boldsymbol{W}_0^{L+1*}\right\|_2.$$

Recalling that $W_i^{L+1} \sim \mathcal{N}(0,1)$, we use Bernstein's inequality to obtain $\mathbb{P}\left[\left\|W^{L+1}\right\|_2 > C\sqrt{n}\right] \leq C'e^{-cn}$, and another application of lemma D.14 gives $\mathbb{P}\left[\mathbb{1}_{\mathcal{E}_{\delta K}} \left\|\boldsymbol{\Gamma}_{\mathcal{J}0}^{L:\ell+2}\boldsymbol{e}_1\right\|_2 > C\right] \leq C''e^{-c'\frac{n}{L}}$ for some constants. Hence after worsening constants

$$\mathbb{P}\left[\mathbb{1}_{\mathcal{E}_{\delta K}} \left\|\boldsymbol{\beta}_{\mathcal{J},0}^{\ell}\right\|_2 > C\sqrt{n}\right] \leq C'e^{-c\frac{n}{L}}. \tag{F.36}$$

We also obtain

$$\mathbb{1}_{\mathcal{E}_{\delta K}} \left\|\tilde{\boldsymbol{\Gamma}}_{\mathcal{J},0}^{\ell:\ell'}\right\| = \mathbb{1}_{\mathcal{E}_{\delta K}} \left\|\boldsymbol{W}_0^{\ell}\boldsymbol{\Gamma}_{\mathcal{J},0}^{\ell-1:\ell'-1}\boldsymbol{P}_{J_{\ell'}}\right\| \underset{a.s.}{\leq} \mathbb{1}_{\mathcal{E}_{\delta K}} \left\|\boldsymbol{W}_0^{\ell}\right\| \left\|\boldsymbol{\Gamma}_{\mathcal{J},0}^{\ell-1:\ell'-1}\right\|$$

$$\mathbb{P}\left[\mathbb{1}_{\mathcal{E}_{\delta K}} \left\|\tilde{\boldsymbol{\Gamma}}_{\mathcal{J},0}^{\ell:\ell'}\right\| > C\sqrt{L}\right] \leq C'e^{-cn} + C''e^{-c'\frac{n}{L}} \leq C'''e^{-c''\frac{n}{L}}$$

where we used an $\varepsilon$-net argument to bound $\left\|\boldsymbol{W}_0^{\ell}\right\|$ and lemma D.14 to bound $\mathcal{E}_{\delta K}\left\|\boldsymbol{\Gamma}_{\mathcal{J},0}^{\ell-1:\ell'-1}\right\|$. We now combine this result with (F.36). It remains to uniformize this result over the choice of $\ell$ and $\mathcal{J}$. Combining (F.26) and (F.34) gives

$$\mathbb{P}\left[\left|\overline{\mathcal{J}}_\eta(\mathcal{M})\right| > C'e^{C''dLn\eta^{2/3}}\right] \leq C'e^{-cd}. \tag{F.37}$$

Denoting the complement of this event by $\mathcal{E}_{\mathcal{J}}$, and setting $\eta = \frac{C_\eta L^{3/2+q}}{\sqrt{n}}$, on this event we have

$$\mathbb{P}\left[\forall \mathcal{J} \in \overline{\mathcal{J}}_\eta(\mathcal{M}), \ell' < \ell \in [L], \quad : \quad \begin{array}{c} \left\{\mathbb{1}_{\mathcal{E}_{\delta K}} \left\|\tilde{\boldsymbol{\Gamma}}_{\mathcal{J},0}^{\ell:\ell'}\right\| \leq C\sqrt{L}\right\} \\ \cap \left\{\mathbb{1}_{\mathcal{E}_{\delta K}} \left\|\boldsymbol{\beta}_{\mathcal{J},0}^{\ell}\right\|_2 \leq C\sqrt{n}\right\} \end{array} \middle| \mathcal{E}_{\mathcal{J}}\right]$$
$$\geq 1 - C'e^{C''dLn\eta^{2/3} - c\frac{n}{L}} \geq 1 - C'e^{C''C_\eta^{2/3}dL^{2+2q/3}n^{2/3} - c\frac{n}{L}} \geq 1 - C'e^{-c'\frac{n}{L}}$$

assuming $n \geq KL^{9+2q}d$ for some constant $K$. Taking a union bound over the probabilities of $\mathcal{E}_{\mathcal{J}}$ or $\mathcal{E}_{K\delta}$ not holding, we finally obtain

$$\mathbb{P}\left[\forall \mathcal{J} \in \overline{\mathcal{J}}_\eta(\mathcal{M}), \ell' < \ell \in [L], \quad : \quad \begin{array}{c} \left\{\left\|\tilde{\boldsymbol{\Gamma}}_{\mathcal{J},0}^{\ell:\ell'}\right\| \leq C\sqrt{L}\right\} \\ \cap \left\{\left\|\boldsymbol{\beta}_{\mathcal{J},0}^{\ell}\right\|_2 \leq C\sqrt{n}\right\} \end{array}\right]$$

$$\geq 1 - C'e^{-c'\frac{n}{L}} - \mathbb{P}\left[\mathcal{E}_{\mathcal{J}}^c\right] - \mathbb{P}\left[\mathcal{E}_{\delta K}^c\right]$$

$$\geq 1 - C'e^{-c'\frac{n}{L}} - C''e^{-c''d} - C'''e^{-c'''d}$$

$$\geq 1 - C''''e^{-c''''d}$$

for appropriate constants, where we used (F.35) to bound $\mathbb{P}\left[\mathcal{E}_{\delta K}^c\right]$, and in the last inequality we used $n \geq KLd$ for some $K$. Combining this with (F.25) and taking a union bound gives the desired result.

ii) We will control $\left\|\boldsymbol{\beta}_{\mathcal{I}_0(\boldsymbol{x}),0}^{\ell} - \boldsymbol{\beta}_{\mathcal{I}_t(\boldsymbol{x}),0}^{\ell}\right\|_2$ using lemma D.21. For $t \in [0, T_\eta]$ we note that by definition of $T_\eta$ and $\overline{\mathcal{J}}_\eta(\boldsymbol{x})$,
$$\mathcal{I}_t(\boldsymbol{x}) \in \overline{\mathcal{J}}_\eta(\boldsymbol{x}).$$

As noted in the previous section, if we set $d$ to satisfy lemma D.8 and $n \geq KdL^{9+2q}$ for some $K$, then the requirements of lemma D.14 are satisfied with

$$\delta_s = C_0 n\eta^{2/3}, \quad K_s = \eta.$$

and

$$\mathbb{P}\left[\mathcal{E}_{\delta K}\right] = \mathbb{P}\left[\mathcal{E}_\delta\right] \geq 1 - Ce^{-cd}$$

where the last bound uses the definition of $\mathcal{E}_\delta$ in (F.35). From the definition of $\overline{\boldsymbol{d}}$ in lemma D.21, on the event $\mathcal{E}_{\delta K}$ we have
$$\left\|\overline{\boldsymbol{d}}\right\|_\infty \leq \delta_s \leq C_0 n\eta^{2/3}.$$

Thus for some fixed $t \in [0, T_\eta]$, if we denote $d_i = |I_{i,t}(\boldsymbol{x}) \ominus I_{i,0}(\boldsymbol{x})|$ for $i \in [L]$, we can apply the second result of lemma D.21, choosing

$$
\begin{aligned}
d_0 &= d \log L, \\
s &= K d_0 L^{2+2q/3} n^{-1/3}, \\
d_b &= K' d_0 L^{2+2q/3} n^{2/3}, \\
s_i &= \frac{K''' d_0 L^{2+2q/3} n^{2/3}}{\max\{1, d_i\}}
\end{aligned} \tag{F.38}
$$

for some appropriately chosen constants $K, K', K'''$. Assuming $n \geq dL^2$ and $n^{1/12}\sqrt{L^{3+2q/3}}d^{1/4} \geq \tilde{K}$ for some constant $\tilde{K}$ to simplify the result, we obtain

$$
\mathbb{P}\left[\mathbb{1}_{\mathcal{E}_{\delta K}} \left\|\boldsymbol{\beta}^\ell_{\mathcal{I}_0(\boldsymbol{x}),0} - \boldsymbol{\beta}^\ell_{\mathcal{I}_t(\boldsymbol{x}),0}\right\|_2 > K'' C_\eta^{2/3} n^{5/12} L^{3+2q/3} d_0^{3/4}\right]
$$

$$
\leq C' e^{-K''' d_0 L^{2+2q/3} n^{2/3}} + C'' e^{-c' \frac{n}{L}}.
$$

The constants $K, K'$ are chosen such that this result can be uniformized over the set of possible supports $\left|\overline{\mathcal{J}}_\eta(\mathcal{M})\right|$ and $[L]$. Since on the event $\mathcal{E}_{\mathcal{J}}$ defined in (F.37) the size of this set is bounded, we have

$$
\mathbb{P}\left[\forall \boldsymbol{x} \in \mathcal{M}, t \in [0, T_\eta], \ell \in [L] : \begin{array}{c} \mathbb{1}_{\mathcal{E}_{\delta K}} \left\|\boldsymbol{\beta}^{\ell-1}_{\mathcal{I}_0(\boldsymbol{x}),0} - \boldsymbol{\beta}^{\ell-1}_{\mathcal{I}_t(\boldsymbol{x}),0}\right\|_2 \\ \leq K'' C_\eta^{2/3} n^{5/12} L^{3+2q/3} d_0^{3/4} \end{array}\middle| \mathcal{E}_{\mathcal{J}}\right] \geq 1 - C' e^{-c d_0 L^{2+2q/3} n^{2/3}}
$$

for some constant $c, C, C'$, assuming $n \geq K'''' L^{9+2q} d$ for some constant $K''''$. Taking a union bound over the complements of $\mathcal{E}_{\mathcal{J}}$ and $\mathcal{E}_{\delta K}$ using (F.37) and (F.35), we have

$$
\mathbb{P}\left[\forall \begin{array}{c} \boldsymbol{x} \in \mathcal{M}, \\ t \in [0, T_\eta], \\ \ell \in [L] \end{array} : \left\|\boldsymbol{\beta}^{\ell-1}_{\mathcal{I}_0(\boldsymbol{x}),0} - \boldsymbol{\beta}^{\ell-1}_{\mathcal{I}_t(\boldsymbol{x}),0}\right\|_2 \leq K'' C_\eta^{2/3} n^{5/12} \log^{3/4}(L) L^{3+2q/3} d^{3/4}\right]
$$

$$
\geq 1 - C' e^{-c d_0 L^{2+2q/3} n^{2/3}} - C'' e^{-c' d}
$$

$$
\geq 1 - C''' e^{-c'' d}
$$

for appropriate constants, assuming $\log(L) L^{2+2q/3} n^{2/3} > \tilde{K}$ for some constant $\tilde{K}$. $\qquad \square$

# G   AUXILIARY RESULTS

**Lemma G.1** (Hoeffding's Inequality (Vershynin, 2018, Theorem 2.2.6)). *Let $X_1, \ldots, X_N$ be independent random variables. Assume that $X_i \in [m_i, M_i]$ for every $i$. Then for any $t > 0$, we have*

$$
\mathbb{P}\left[\sum_{i=1}^N (X_i - \mathbb{E}[X_i]) \geq t\right] \leq \exp\left(-\frac{2t^2}{\sum_{i=1}^N (M_i - m_i)^2}\right).
$$

**Lemma G.2** (Bernstein's inequality (Vershynin, 2018, Theorem 2.8.1)). *Let $X_1, \ldots, X_N$ be independent mean-zero subexponential random variables. Then, for every $t \geq 0$, one has*

$$
\mathbb{P}\left[\left|\sum_{i=1}^N X_i\right| \geq t\right] \leq 2 \exp\left(-c \min\left\{\frac{t^2}{\sum_{i=1}^N \|X_i\|_{\psi_1}^2}, \frac{t}{\max_i \|X_i\|_{\psi_1}}\right\}\right),
$$

*where $c > 0$ is an absolute constant, and $\|\cdot\|_{\psi_1} = \inf\{t > 0 \mid \mathbb{E}[e^{|\cdot|/t}] \leq 2\}$ is the subexponential norm.*

**Lemma G.3** (Bernstein's inequality for bounded RVs - (Vershynin, 2018) Thm. 2.8.4). *For $X_1, \ldots, X_n$ independent, zero mean random variables such that $\forall i : |X_i| < K$, and every $t \geq 0$, we have*

$$
\mathbb{P}\left[\left|\sum_{i=1}^n X_i\right| \geq t\right] \leq 2 \exp\left(-\frac{t^2/2}{\sigma^2 + Kt/3}\right)
$$

*where $\sigma^2 = \sum_{i=1}^n \mathbb{E} X_i^2$.*

**Lemma G.4** (Hanson-Wright Inequality (Vershynin, 2018, Theorem 6.2.1)). *Let $\boldsymbol{g}$ be a vector of $n$ i.i.d., mean zero, sub-Gaussian variables and $\boldsymbol{A}$ be an $n \times n$ matrix. Then for any $t > 0$, we have*

$$\mathbb{P}\left[|\boldsymbol{g}^* \boldsymbol{A} \boldsymbol{g} - \mathbb{E}\boldsymbol{g}^* \boldsymbol{A} \boldsymbol{g}| \geq t\right] \leq 2\exp\left(-c\min\left\{\frac{t^2}{K^4 \|\boldsymbol{A}\|_F^2}, \frac{t}{K^2 \|\boldsymbol{A}\|}\right\}\right)$$

*where $\max_i \|g_i\|_{\psi_2} \leq K$ (with $\|\cdot\|_{\psi_2}$ denoting the sub-Gaussian norm).*

**Lemma G.5** (Freedman's Inequality (Freedman, 1975, Theorem 1.6)). *Let $(\Delta^i, \mathcal{F}^i)$ be a sequence of martingale differences, with*

$$\mathbb{E}\left[\Delta^i \mid \mathcal{F}^{i-1}\right] = 0,$$

*and suppose that*

$$|\Delta^i| \leq R \quad a.s..$$

*Define the quadratic variation*

$$V^L = \sum_{i=1}^{L} \mathbb{E}\left[\left(\Delta^i\right)^2 \mid \mathcal{F}^{i-1}\right].$$

*Then*

$$\mathbb{P}\left[\exists i = 1 \ldots L \text{ s.t. } \left|\sum_{\ell=1}^{i} \Delta^\ell\right| > t \quad and \quad V^i \leq \sigma^2\right] \leq 2\exp\left(-\frac{t^2/2}{\sigma^2 + Rt/3}\right).$$

**Lemma G.6** (Moment control Freedman's (de la Peña, 1999)). *Let $(\Delta^i, \mathcal{F}^i)$ be a sequence of martingale differences, with*

$$\mathbb{E}\left[\Delta^i \mid \mathcal{F}^{i-1}\right] = 0,$$

*and suppose that*

$$\mathbb{E}\left[\left(\Delta^i\right)^k \mid \mathcal{F}^{i-1}\right] \leq \frac{k!}{2}\mathbb{E}\left[\left(\Delta^i\right)^2 \mid \mathcal{F}^{i-1}\right] R^{k-2} \qquad \forall k, \ a.s..$$

*Set*

$$V^j = \sum_{i=1}^{j} \mathbb{E}\left[\left(\Delta^i\right)^2 \mid \mathcal{F}^{i-1}\right].$$

*Then*

$$\mathbb{P}\left[\exists i = 1 \ldots j \text{ s.t. } \left|\sum_{\ell=1}^{i} \Delta^\ell\right| > t \quad and \quad V^i \leq \sigma^2\right] \leq 2\exp\left(-\frac{t^2/2}{\sigma^2 + Rt}\right).$$

**Lemma G.7** (Martingales with subgaussian increments). *Let $(\Delta^i, \mathcal{F}^i)$ be a sequence of martingale differences, and suppose that*

$$\mathbb{E}\left[\exp\left(\lambda\Delta^i\right) \mid \mathcal{F}^{i-1}\right] \leq \exp\left(\frac{\lambda^2 V^2}{2}\right), \ \forall \lambda, \ a.s.$$

*Then*

$$\mathbb{P}\left[\left|\sum_{i=1}^{L} \Delta^i\right| > t\right] \leq 2\exp\left(-\frac{t^2}{2LV^2}\right).$$

*Proof.* By assumption, $\mathbb{E}[\Delta^i] = 0$ for each $i \in [L]$. We calculate using standard properties of the conditional expectation

$$\mathbb{E}\left[e^{\lambda \sum_{i=1}^{L} \Delta^i}\right] = \mathbb{E}\left[\mathbb{E}\left[e^{\lambda \sum_{i=1}^{L} \Delta^i} \mid \mathcal{F}^{L-1}\right]\right]$$
$$= \mathbb{E}\left[e^{\lambda \sum_{i=1}^{L-1} \Delta^i}\mathbb{E}\left[e^{\lambda\Delta^L} \mid \mathcal{F}^{L-1}\right]\right] \leq e^{\lambda^2 V^2/2}\mathbb{E}\left[e^{\lambda \sum_{i=1}^{L-1} \Delta^i}\right].$$

Moreover, one has $\mathbb{E}[e^{\lambda\Delta^1} \mid \mathcal{F}^0] = \mathbb{E}[e^{\lambda\Delta^1}] \leq e^{\lambda^2 V^2/2}$. An induction therefore implies

$$\mathbb{E}\left[e^{\lambda \sum_{i=1}^{L} \Delta^i}\right] \leq e^{\lambda^2 LV^2/2},$$

and the result follows from standard equivalence properties of subgaussian random variables (Vershynin, 2018, Proposition 2.5.2). □

**Lemma G.8** (Azuma-Hoeffding Inequality (Azuma, 1967)). *Let $(\Delta^i, \mathcal{F}^i)$ be a sequence of martingale differences, and suppose that*

$$\left|\Delta^i\right| \leq R_i \text{ a.s..}$$

*Then*

$$\mathbb{P}\left[\left|\sum_{\ell=1}^{L}\Delta^i\right| > t\right] \leq 2\exp\left(\frac{-t^2}{2\sum\limits_{\ell=1}^{L}R_i^2}\right).$$

**Lemma G.9** (Chi and Inverse-Chi Expectations). *Let $X \sim \chi(n)$ be a chi random variable with $n$ degrees of freedom, equal to the square root of the sum of $n$ independent and identically distributed squared $\mathcal{N}(0,1)$ random variables. Then*

$$\mathbb{E}[X] = \sqrt{2}\frac{\Gamma(\frac{1}{2}(n+1))}{\Gamma(\frac{1}{2}n)},$$

*and, if $n \geq 2$,*

$$\mathbb{E}[X^{-1}] = \frac{1}{\sqrt{2}}\frac{\Gamma(\frac{1}{2}(n-1))}{\Gamma(\frac{1}{2}n)}.$$

*Proof.* We use the fact that the density of $X$ is given by

$$\rho(x) = \mathbb{1}_{x \geq 0}(x)\frac{1}{2^{n/2-1}\Gamma(\frac{1}{2}n)}x^{n-1}e^{-x^2/2},$$

which can be proved easily using the Gaussian law and a transformation to spherical polar coordinates (Muirhead, 1982, Theorem 2.1.3). The expectation of $X$ then results from a simple sequence of calculations using the change of variables formula:

$$\begin{aligned}
\mathbb{E}[X] &= \frac{2}{2^{n/2}\Gamma(\frac{1}{2}n)}\int_0^{\infty} x^n e^{-x^2/2}\,\mathrm{d}x \\
&= \frac{1}{2^{n/2}\Gamma(\frac{1}{2}n)}\int_0^{\infty} x^{n/2-1/2}e^{-x/2}\,\mathrm{d}x \\
&= \frac{\sqrt{2}}{\Gamma(\frac{1}{2}n)}\int_0^{\infty} x^{(n/2+1/2)-1}e^{-x}\,\mathrm{d}x \\
&= \sqrt{2}\frac{\Gamma(\frac{1}{2}(n+1))}{\Gamma(\frac{1}{2}n)}.
\end{aligned}$$

Now we study $X^{-1}$. By the change of variables formula, its density is given by

$$\rho'(x) = \mathbb{1}_{x \geq 0}(x)\frac{1}{2^{n/2-1}\Gamma(\frac{1}{2}n)}x^{-n}e^{-1/(2x^2)}.$$

A similar sequence of calculations then yields

$$\begin{aligned}
\mathbb{E}[X^{-1}] &= \frac{2}{2^{n/2}\Gamma(\frac{1}{2}n)}\int_0^{\infty} x^{-n}e^{-1/(2x^2)}\,\mathrm{d}x \\
&= \frac{1}{2^{n/2}\Gamma(\frac{1}{2}n)}\int_0^{\infty} x^{-\frac{1}{2}(n+1)}e^{-1/(2x)}\,\mathrm{d}x \\
&= \frac{1}{2^{n/2}\Gamma(\frac{1}{2}n)}\int_0^{\infty} x^{\frac{1}{2}(n-1)-1}e^{-\frac{1}{2}x}\,\mathrm{d}x \\
&= \frac{1}{\sqrt{2}\Gamma(\frac{1}{2}n)}\int_0^{\infty} x^{\frac{1}{2}(n-1)-1}e^{-x}\,\mathrm{d}x \\
&= \frac{1}{\sqrt{2}}\frac{\Gamma(\frac{1}{2}(n-1))}{\Gamma(\frac{1}{2}n)},
\end{aligned}$$

provided $n > 1$. $\qquad\square$

**Lemma G.10** (Equivalence of $\ell^p$ Norms). *Let $1 \leq p \leq q \leq +\infty$. Then for every $x \in \mathbb{R}^n$ one has*

$$\|x\|_q \leq \|x\|_p \leq n^{1/p-1/q}\|x\|_q.$$

**Lemma G.11** (Gaussian Moments). *Let $p \geq 1$, and let $g \sim \mathcal{N}(0,1)$ be a standard normal random variable. Then*

$$\mathbb{E}[|g|^p] = 2^{p/2}\frac{\Gamma\left(\frac{p+1}{2}\right)}{\Gamma\left(\frac{1}{2}\right)}; \qquad \mathbb{E}\big[[g]_+^p\big] = \frac{1}{2}\mathbb{E}[|g|^p],$$

*where $[x]_+ = \max\{x, 0\}$. In particular $\mathbb{E}[|g|^p] \leq p^{p/2}$, so that $g$ is subgaussian and $g^2$ is subexponential.*

