# OpenReview forum: "Deep Networks and the Multiple Manifold Problem"
_ICLR.cc/2021/Conference — ICLR 2021 Poster_

### Official Review · AnonReviewer1 · 2020-10-30
**Important result, difficult to verify the proof, possibly ICLR is not a proper venue**

**Rating:** 6
**Confidence:** 2

**Review:**

The authors consider a binary classification task. As a model the authors use a deep fully-connected neural network and train it to separate the submanifolds, representing different classes. They assume that sub-manifolds belong to the unit sphere. Also, the authors restrict their analysis to a one-dimensional case. The main claim is that by increasing depth we can improve model generalization of a network, trained by SGD.

The proved result seems to be imporant for understanding the generalization ability of deep neural networks. While proving it the authors used various imporant tools from martingale concentration approach.

However, the paper is not well suited for such venue as ICLR. Moreover, I am sure that the proof itself contains some parts that are of separate interest for the community. I would propose to divide the paper into several parts, so that some of them contain general results about approaches to prove concentration inequalities, which are of general interest and can be used somewhere else. Also, I would expect more comments on which tools, used for proving the main results, are completely new and how they can be used to establish similar results for other network architectures.
Also, I would expect some comments how in principle the restriction that the submanifolds belong to the unit sphere can be removed. Some more discussion how the proposed setup is related to the setup, considered in the paper of Goldt (2019), are needed.

========

After reading authors comments.

In principle, I tried to understand the main steps of the proof, they looks OK. Although, I can not verify the details of the proof.
I still think that ICLR venue is not OK for such long submissions.
In principle, I am OK to increase the grade for one point.

---

> ### Author Response · Authors · 2020-11-14
> **Response to AnonReviewer1**
>
> **Generally applicable tools/approaches + similar results for other architectures:** We thank the reviewer for suggesting that we highlight the novel contributions of this work that we believe will be more generally applicable or interesting to a wider audience. Due to the importance of this point, we address it in the general response above.
>
> **Removing the unit sphere restriction:** While the data model considered has limitations, normalization of data in neural network training is common in practice, and simplifies the analysis since the neural tangent kernel governing the dynamics can be expressed in terms of a translationally invariant operator on the sphere. We are actively pursuing generalizations of the certificate construction argument to more general curves.
>
> **Comparison to Goldt (2019):** If accepted, we will utilize additional space in the related work section for a more in-depth comparison to the hidden manifold model of Goldt et al. The motivations for our data model are very similar to those of the hidden manifold model, and the hidden manifold model adds some desirable features not present in our formulation of the multiple manifold problem in this paper, including modeling statistical variations in the data around the base manifold with some noise. We believe the nonlinearity present in the hidden manifold model, namely a nonlinear function acting componentwise on the data (which are isotropic gaussian samples from a low-dimensional subspace), restricts the class of interesting geometries that can be modeled relative to the problem we consider, where the data are allowed to (in the higher-dimensional problem formulation) be drawn from a density supported on an arbitrary Riemannian manifold. If the nonlinear function in the hidden manifold model was not restricted to act componentwise, the resulting model could have additional generality relative to ours (for example, if $f$ was only required to be Lipschitz). The analysis of these two models also differs fundamentally: Goldt et al. pursue an asymptotic mean-field analysis of a two-layer neural network training which is shown empirically to give accurate predictions for actual systems in certain regimes, whereas our analysis is nonasymptotic in nature, and for networks of arbitrary depth.

---

### Official Review · AnonReviewer2 · 2020-10-30
**Rigorous and well-written paper**

**Rating:** 7
**Confidence:** 4

**Review:**

The paper studies the conditions for a deep fully-connected network to separate low-dimensional data classes. A binary classification setting is considered, where the two classes are modelled as two different manifolds. The manifolds are assumed to be one-dimensional for the ease of analysis. It is shown that the network depth should be sufficiently large so as to adapt to the geometrical properties of data (e.g. the manifold curvature); the network width should increase polynomially with the network depth; and the number of data samples should also scale at a polynomial rate with the network depth. The authors show that if these conditions are met, with high probability a randomly initialized network converges to a classifier that separates the two class manifolds. The proof technique relies on conditioning the network parameters of the l-th layer on the parameters of the previous layers using a Martingale model, which gives sharp concentration guarantees.

The paper is rigorous and well-written, and very detailed proofs have been provided for the presented results. The authors may still find it useful to address the following issues:

1. The separation Delta between the two manifolds seems to be a critical parameter in the proposed performance bounds. However, the definition of Delta in Section 3.1 is not very clear. In particular, what does the notation \angle(x, x') mean? Does this mean Euclidean distance, or angle as the notation suggests? If this means an angle, please explain whether this refers to the angle of points on the sphere, or the angle measured on the tangent space of the Riemannian manifold, etc.

2. Although I like very much the analysis proposed by the authors, some of the main assumptions regarding the studied setting may be too restrictive in practice:

a. The proposed bounds apply only when the separation parameter Delta is positive. In a practical application, the manifolds representing different classes may be (asymptotically) intersecting due to the degeneracy of image transformations/articulations in some manifold regions - e.g. extreme illumination conditions, scale changes, etc.

b. The assumption that the data samples lie exactly on a Riemannian manifold also seems to be restrictive. A probabilistic model for the data samples, e.g. involving a noise model capturing the deviations of the points from the manifold, would have been more realistic. Would the theoretical analysis be completely intractable under such extensions, or could that be a potential future direction of the study?

It would be nice if the authors could comment on such issues.

---

> ### Author Response · Authors · 2020-11-14
> **Response to AnonReviewer2**
>
> **The inter-manifold distance $\Delta$:** As the reviewer notes, this is an important parameter in the problem since in some sense it sets the “difficulty” of the classification problem. We define it (and $\angle(\boldsymbol{x},\boldsymbol{x}’)$ whenever this notation appears) as the spherical distance $\cos^{-1}( \langle \boldsymbol{x}, \boldsymbol{x}’\rangle )$, where the inner product is the usual euclidean one. We denote the intrinsic distance on the data manifolds by $\mathrm{dist}_{\mathcal{M}}$; but since we restrict our attention to regular curves, in these cases we typically just work with a parameterization with respect to arc length instead.
>
> **Probabilistic model for data:** We believe it is possible to obtain results for noisy versions of the problem studied in our work using essentially the machinery of this paper, in a setting where the noise level is small. The problem of extending to settings where the noise level is larger is of significant practical and theoretical importance: indeed, we are currently studying this setting as motivated by an application to gravitational wave astronomy. Here, it seems necessary to consider a statistical model for the noise in order to avoid having to control the NTK uniformly over a “tube” in the ambient space around the manifold (which would incur rates exponential in the ambient dimension, at best). In addition to technical issues for concentration along these lines, there are modeling questions one must consider as well; for example, it may make more sense in this setting to measure performance in terms of an alternate metric, like minimum false positive rate at a specified false negative rate, which could change the nature of the analysis required.
>
> **Intersecting data manifolds:** We agree with the reviewer that there are settings in practical imaging applications where perfect separation is not achievable (for example, as the reviewer suggests, under extreme illuminations). We agree that it would be necessary to modify our model problem in such settings to address this. In some cases, these modifications may be made naturally: illustrating in the context of the illumination example, these kinds of intersections typically only occur with images under low-light conditions, so it is possible to incorporate a lower bound on the illumination level into the model to avert this issue. On the other hand, we believe that our analytical approach (in particular, the approach to dynamics via certificates) has value in addressing these more challenging scenarios, as well.

---

### Official Review · AnonReviewer4 · 2020-10-30
**hard to follow and too long**

**Rating:** 5
**Confidence:** 1

**Review:**

The paper analyzes deep networks in terms of multiple manifold problems. However, it needs easier explanations for a wider audience.  Moreover, I think the article was written as a book rather than a conference paper, considering 245 pages.

What would be the actual benefits of the analysis?

The paper talks about one-dimensional cases, and I was wondering if the multiple manifold problem in one-dimension is enough. If I understand correctly, this one-dimension manifold is supposed to represent one class in the last layer.

Some notations are hard to follow, especially in Section 3.1. Actually, some of them are used without the description.)

---

> ### Author Response · Authors · 2020-11-14
> **Response to AnonReviewer4**
>
> **Benefits of the analysis:** We discuss our vision for the analysis in the introduction to the paper: we see a need for theory and model problems that capture specific structures, like low-dimensional data, that are present in practical applications where deep learning achieves strong performance. Our results span the entire “pipeline” of training a deep network to classify structured data, going from the data model to a generalization guarantee (Theorem 1), and our results give some qualitative insights into the roles of architectural hyperparameters like the network depth and width in solving the classification problem. On a more technical level, in our general rebuttal we have highlighted a number of concrete benefits and novelties of our analysis.
>
> **Higher dimensional manifolds:** The manifolds in our problem are defined in the input space, and represent classes of inputs with different labels. A natural direction of future work is to consider higher dimensional manifolds. Considering the motivating example of image articulation manifolds for example, one would typically expect the dimension of the manifold to roughly correspond to the dimension of the group of transformation applied to the image that will leave the label invariant.
>
> Note that some parts of our result, such as the uniform concentration result for the NTK (Theorem 2), are already applicable to a higher-dimensional setting, and Theorem 1 can be generalized to this setting with suitable extensions of Lemmas B.13 and C.6. It will be natural to expect an exponential dependence of the sample complexity on the dimension of the manifold in this case, and perhaps other parameters such as the depth will incur such an exponential dependence as well.

---

### Official Review · AnonReviewer3 · 2020-10-31
**Review of "Deep Networks and the Multiple Manifold Problem"**

**Rating:** 8
**Confidence:** 2

**Review:**

The paper under review studies the question of whether gradient descent can solve the problem of calibrating a deep neural network for separating two submanifolds of the sphere. The problem studied in the paper is very interesting and as been the subject of recent increasing interest in the machine learning community. The contribution is restricted to a simple set up and addresses the question in the finite sample regime. The framework of the analysis hinges on the Neural Tangent Kernel approximation of Jacot et al.

This extremely technical paper is indimidating and I wonder how many readers will actually read the proofs. Moreover, one may wonder if the NTK approach precludes a deeper understanding of the actual performance of the network under consideration.

---

> ### Author Response · Authors · 2020-11-14
> **Response to AnonReviewer3**
>
> **Training beyond the NTK regime:** We agree with the reviewer that the NTK regime may not capture certain interesting aspects of learning, and the scaling required to stay in this regime during the entire training process leads to unrealistically wide networks. Any improved understanding of changes in the NTK during training have the potential to lead to improved scaling in our result. In its current form, the analysis in the NTK regime can still be useful in understanding early stages of training (if training with a small learning rate). Our results also contribute to the understanding of the separation between the NTK regime and other regimes where the network parameters can change considerably. In particular, we show that training in the NTK regime can be sufficient to solve certain tasks if the data contains low dimensional structure, without requiring a number of samples that scales polynomially with the ambient dimension.

---

### Author Response · Authors · 2020-11-14
**Response to Reviewers (1/2)**

We thank the reviewers for their insightful comments, and for their effort in reviewing what we agree is a fairly technical paper with a long supplementary appendix. The broad motivation of our work is to understand how neural networks leverage low-dimensional structure in datasets. Since most datasets in practical learning problems exhibit some form of low-dimensional structure, we believe that this broad goal is highly relevant to the ICLR community, as is the setting for our analysis. We also believe that the tools and approaches that our paper introduces can be brought to bear on other analytical problems of interest to the ICLR community.

Following the advice of the reviewers, if accepted, in the final version of the paper we will highlight these novel aspects, which include:
- **Uniform concentration of the neural tangent kernel for deep networks:** Our sharp concentration result for the NTK (Theorem 2) relies essentially on the compositional structure of the network and the fact that the weights are subgaussian, which allows the application of martingale concentration results. As we discuss in section 4.1, it is this property that leads us to believe that our argument for optimal rates of concentration can be extended to more general network architectures, albeit with additional technicalities in handling more structured maps such as convolutions.
At a more technical level, we can highlight the novelty of the “uniform” nature of Theorem 2: it controls the worst fluctuations in the NTK over the entire data manifold.  Most previous results on concentration of the NTK consider only pointwise concentration (e.g. [1], [2]; analogous to our Theorem B.3). Obtaining uniform concentration for the NTK of a _deep_ neural network consumes a good deal of effort in our proofs (appendix D.3), but it is certainly necessary in our setting, where generalization is the aim.
- **Control of the Lipschitz properties of ReLU networks:** We show in Lemma D.12 that when the domain is a single curve satisfying our regularity assumptions, the Lipschitz constant of a ReLU network at initialization is bounded by a constant with constant probability. In particular, it is independent of the ambient dimension, depth and width. We also control the Lipschitz constant during training, showing that it scales polynomially with the depth. Such results may be of general interest in understanding the robustness of neural networks, and we anticipate generalizing them to input manifolds of arbitrary dimension in future work.
- **Analysis of training via certificate construction:** We believe that connections we make between the data geometry, the decay properties of the neural tangent kernel, and the depth of the network are novel. They highlight the role of depth as a “fitting resource” in the problem. This approach can potentially be generalized to analyse the dynamics in other overparameterized architectures, and may lead to novel relationships between the data geometry and the architecture. In convolutional networks for instance, we expect the resulting kernel to depend on cross-correlations between inputs as opposed to angles.
- **Sample complexity requirements with structured data:** The sample complexity required to guarantee generalization in our setting (Theorem 1) depends only on the depth of the network. This is in distinction to similar bounds that are based on minimum description length of the classifier, which may be significantly looser. The precise exponent on the depth in this theorem is suboptimal, but it appears to be at least somewhat nontrivial to obtain (Lemma B.6) results of this type that avoid an unfortunate dependence on the network width or a superpolynomial dependence on the depth--we require an argument that carefully controls the evolution of the Lipschitz constant of the network throughout training (as alluded to above) by connecting it to the NTK. These issues do not appear to be present in the case of, e.g., a one hidden layer NTK, and the approach we develop here may be used as a basis to develop sharper rates in future work.
- **Generalization bounds for deep networks:** For deep overparameterized networks, there are a number of results on convergence of gradient descent to a global minimum of the training loss. To our knowledge however, there are few generalization bounds for deep networks trained efficiently with gradient descent. These usually apply to data models that do not exhibit the low-dimensional structure common in many applications. Our guarantee does not depend on any property of the trained network (such as norms of weights) that is not readily available before training. This highlights the benefits of considering structured data, which is not always a salient feature of theoretical results on neural networks.

We believe that highlighting these points will also help the reader to access the novel and generally applicable aspects of the analysis, despite the length of the paper.

---

### Author Response · Authors · 2020-11-14
**Response to Reviewers (2/2)**

**Notation:** We apologize for a lack of a unified set of definitions for the notation employed in the main body of the paper. A complete account of the notation used in the paper is given in the appendices, in section A.3.1. If accepted, we will utilize additional space in the main body of the paper to add an extra subsection following the related work that defines essential notation (for example, the definition of the spherical distance and notions from functional analysis).


References.

[1] Arora et al., On Exact Computation with an Infinitely Wide Neural Net. http://arxiv.org/abs/1904.11955

[2] Allen-Zhu et al., A Convergence Theory for Deep Learning via Over-Parameterization. http://arxiv.org/abs/1811.03962

---

### Decision · Program_Chairs · 2021-01-07
**Final Decision**

**Decision:**

Accept (Poster)

**Comment:**

This paper introduces the multiple manifold problem - in a simple setting there are two data manifolds representing the positive and negative samples, and the goal is to train a neural network (or any predictor) that separates these two manifolds. The paper showed that this is possible to do with a deep neural network under certain assumptions - notably on the shape of the manifold and also on the ability of the neural network to represent certain functions (which is harder to verify, and only verified for a 1-d case in the paper). The optimization of neural network falls in the NTK regime but requires new techniques. Overall the question seems very natural and the results are reasonable first steps. There are some concerns about clarity that the authors should address in the paper.